The phylogenetic relationships of basal archosauromorphs, with an emphasis on the systematics of proterosuchian archosauriforms

Ezcurra Martín D. martindezcurra@yahoo.com.ar MDE340@bham.ac.uk
School of Geography, Earth and Environmental Sciences, University of Birmingham , Birmingham , United Kingdom
Sección Paleontología de Vertebrados, Museo Argentino de Ciencias Naturales , Buenos Aires , Argentina
GeoBio-Center, Ludwig-Maximilians-Universität München , Munich , Germany
Sues Hans-Dieter
Electronic publication date: 2016 Apr 28
Publication date: 2016
Volume: 4
Electronic Location ID: e1778
Received 2015 Dec 1; Accepted 2016 Feb 18
Copyright: ©2016 Ezcurra
Copyright year: 2016
Copyright holder: Ezcurra
License: This is an open access article distributed under the terms of the Creative Commons Attribution License, which permits unrestricted use, distribution, reproduction and adaptation in any medium and for any purpose provided that it is properly attributed. For attribution, the original author(s), title, publication source (PeerJ) and either DOI or URL of the article must be cited.
License URL: https://creativecommons.org/licenses/by/4.0/

Keywords: Diapsida, Archosauromorpha, Permian, Cladistics, Archosauria, Phylogeny, Triassic, Macroevolution

Funding: Emmy Noether Programme of the Deutsche Forschungsgemeinschaft BU 2587/3-1 College of Life and Environmental Sciences (University of Birmingham) Marie Curie Career Integration Grant PCIG14-GA-2013-630123 National Geographic Young Explorers Grant #9467-14 Synthesis Program Doris and Samuel P. Welles Fund Jurassic Foundation Society of Vertebrate Paleontology This research was primarily supported by the Emmy Noether Programme of the Deutsche Forschungsgemeinschaft (BU 2587/3-1 to Richard Butler), with additional support from the College of Life and Environmental Sciences (University of Birmingham), a Marie Curie Career Integration Grant (PCIG14-GA-2013-630123 to Richard Butler), and a National Geographic Young Explorers Grant (#9467-14). The Synthesis Program funded his visit to the collection of vertebrate palaeontology of the Museum national d’Historie Neturelle in Paris, and the Doris and Samuel P. Welles Fund funded his visit to the University of California, Museum of Paleontology in Berkeley. Grants from the Jurassic Foundation and the Society of Vertebrate Paleontology (Jackson School of Geosciences Student Member Travel Grant) also contributed to this research. The funders had no role in study design, data collection and analysis, decision to publish, or preparation of the manuscript.

==============================
The early evolution of archosauromorphs during the Permo-Triassic constitutes an excellent empirical case study to shed light on evolutionary radiations in deep time and the timing and processes of recovery of terrestrial faunas after a mass extinction. However, macroevolutionary studies of early archosauromorphs are currently limited by poor knowledge of their phylogenetic relationships. In particular, one of the main early archosauromorph groups that need an exhaustive phylogenetic study is “Proterosuchia,” which as historically conceived includes members of both Proterosuchidae and Erythrosuchidae. A new data matrix composed of 96 separate taxa (several of them not included in a quantitative phylogenetic analysis before) and 600 osteological characters was assembled and analysed to generate a comprehensive higher-level phylogenetic hypothesis of basal archosauromorphs and shed light on the species-level interrelationships of taxa historically identified as proterosuchian archosauriforms. The results of the analysis using maximum parsimony include a polyphyletic “Prolacertiformes” and “Protorosauria,” in which the Permian Aenigmastropheus and Protorosaurus are the most basal archosauromorphs. The enigmatic choristoderans are either found as the sister-taxa of all other lepidosauromorphs or archosauromorphs, but consistently placed within Sauria. Prolacertids, rhynchosaurs, allokotosaurians and tanystropheids are the major successive sister clades of Archosauriformes. The Early Triassic Tasmaniosaurus is recovered as the sister-taxon of Archosauriformes. Proterosuchidae is unambiguosly restricted to five species that occur immediately after and before the Permo-Triassic boundary, thus implying that they are a short-lived “disaster” clade. Erythrosuchidae is composed of eight nominal species that occur during the Early and Middle Triassic. “Proterosuchia” is polyphyletic, in which erythrosuchids are more closely related to Euparkeria and more crownward archosauriforms than to proterosuchids, and several species are found widespread along the archosauromorph tree, some being nested within Archosauria (e.g., “Chasmatosaurus ultimus,” Youngosuchus). Doswelliids and proterochampsids are recovered as more closely related to each other than to other archosauromorphs, forming a large clade (Proterochampsia) of semi-aquatic to aquatic forms that includes the bizarre genus Vancleavea. Euparkeria is one of the sister-taxa of the clade composed of proterochampsians and archosaurs. The putative Indian archosaur Yarasuchus is recovered in a polytomy with Euparkeria and more crownward archosauriforms, and as more closely related to the Russian Dongusuchus than to other species. Phytosaurs are recovered as the sister-taxa of all other pseudosuchians, thus being nested within Archosauria.

Introduction

The early evolution of the archosauromorphs during the Triassic is an excellent example of an adaptative radiation in the fossil record (Brusatte et al., 2008; Nesbitt, 2011). In the aftermath of the Permo-Triassic mass extinction, multiple, anatomically well diversified archosauromorph groups appear for the first time in the fossil record, including semi aquatic or entirely aquatic forms (e.g., tanystropheids, doswelliids, proterochampsids, some poposauroids), highly specialized herbivores (e.g., allokotosaurians, rhynchosaurs), and massive predators (e.g., erythrosuchids, “rauisuchians”). As a result, the early evolution of archosauromorphs constitutes an excellent empirical case study to shed light on evolutionary radiations in deep time and the timing and processes of recovery of terrestrial faunas after a mass extinction. However, macroevolutionary studies of early archosauromorphs are substantially limited by poor knowledge of their phylogenetic relationships (Ezcurra, Butler & Gower, 2013). Many early archosauromorph species have not been previously included in a quantitative phylogenetic analysis, and have been historically included within groups that are probably non-monophyletic as often conceived (e.g., “Prolacertiformes,” Proterosuchidae; Dilkes, 1998; Modesto & Sues, 2004; Gottmann-Quesada & Sander, 2009; Ezcurra, Lecuona & Martinelli, 2010; Ezcurra, Butler & Gower, 2013; Ezcurra, Scheyer & Butler, 2014). In addition, the higher-level phylogenetic relationships of the main lineages of archosauromorphs are highly contentious and there is limited consensus between the results recovered by different studies (e.g., Dilkes, 1998; Modesto & Sues, 2004; Gottmann-Quesada & Sander, 2009; Ezcurra, Scheyer & Butler, 2014; Pritchard et al., 2015).

One of the main early archosauromorph groups that need an exhaustive phylogenetic study is “Proterosuchia,” which, as historically conceived, includes members of both Proterosuchidae and Erythrosuchidae (Reig, 1970; Charig & Reig, 1970; Charig & Sues, 1976; Ezcurra, Butler & Gower, 2013). Indeed, most proterosuchian species have not yet been included in quantitative phylogenetic analyses, and their phylogenetic positions among basal archosauriforms or within either Proterosuchidae or Erythrosuchidae is in state of flux (e.g., Guchengosuchus shiguaiensis, Cuyosuchus rusconi, Chalishevia cothurnata, Shansisuchus kuyeheensis, Garjainia madiba, “Chasmatosaurus” yuani, “Blomosuchus georgii,” Vonhuenia friedrichi, Chasmatosuchus rossicus, Tasmaniosaurus triassicus, Kalisuchus rewanensis). The proterosuchians represent the most basal known archosauriforms, and, as a result, an understanding of their phylogenetic relationships is crucial to attempts to reconstruct the interrelationships of more crownward archosauriforms and the early evolutionary history of Archosauriformes as a whole. However, the poor current phylogenetic understanding of the proterosuchians hampers the development of diagnoses for Proterosuchidae and Erythrosuchidae, and the taxonomic inclusiveness of these clades remains uncertain (Ezcurra, Butler & Gower, 2013). This contribution focuses on the phylogenetic relationships of non-archosaurian archosauromorphs, with a special emphasis on the interrelationships among taxa historically identified as proterosuchian archosauriforms.

Previous work

In the following pages I discuss the work conducted by previous authors on the higher-level phylogenetic relationships of Archosauromorpha, with emphasis in the historical background of the phylogenetic interrelationships of the “Proterosuchia.” The cladistic history of the relationships among non-proterosuchian archosauriforms has been recently discussed by Nesbitt (2011) and, as a result, is not summarized here.

The higher-level phylogenetic relationships of early archosauromorphs

During the pre-cladistic era, small and gracile Permo-Triassic diapsids (including some early archosauromorphs) were frequently included within the Order “Eosuchia,” which was suggested to have given rise to lepidosauromorphs, archosaurs and even some marine reptiles, such as plesiosaurs (e.g., Romer, 1956; Romer, 1966; Benton, 1982). However, the classification of diapsid reptiles was chaotic and in state of flux prior to the advent of quantitative cladistic analyses. The first cladistic analyses that focused on the higher-level relationships of archosauromorphs were conducted by Benton (1984a), Benton (1985) and Evans (1984). Benton (1984a) and Benton (1985) recovered four main clades within Archosauromorpha, namely Pterosauria, Rhynchosauria, “Prolacertiformes” and Archosauria (Fig. 1A). Pterosauria represented the earliest branching archosauromorphs, and Trilophosaurus and Rhynchosauria were successive sister-taxa of a clade composed of “Prolacertiformes” and Archosauria (Benton, 1985). “Prolacertiformes” included Permo-Triassic long-necked archosauromorphs (e.g., Protorosaurus, Prolacerta and tanystropheids), and archosaurs included more bulky forms, such as Erythrosuchus, Euparkeria, and dinosaur and crocodile precursors (thus being largely equivalent to the current concept of Archosauriformes). In particular, Benton (1985) was uncertain whether the South African archosauromorph Proterosuchus represented a prolacertiform or an archosaur. The analysis of Evans (1984) and subsequent analyses (e.g., Gauthier, Kluge & Rowe, 1988; Chatterjee, 1986; Bennett, 1996) also placed Rhynchosauria as the sister-taxon of a clade composed of “Prolacertiformes”/“Protorosauria” and Archosauria (Figs. 1C–1E). In addition, Evans (1988), Evans (1990) and Gauthier, Kluge & Rowe (1988) tentatively added other main lineages to Archosauromorpha, namely the aquatic choristoderans and thalattosaurians, and the gliding kuehneosaurids (Figs. 1B and 1D). As a result, the pioneering cladistic analyses conducted during the 1980s largely agreed in the recognition of three clades of archosauromorphs (i.e., “Prolacertiformes,” Rhynchosauria, and Archosauriformes) and the basal position of rhynchosaurs with respect to prolacertiforms and archosauriforms, a result also recovered by some analyses during the subsequent decade (e.g., Bennett, 1996). However, a point of disagreement between these early analyses was the position of Trilophosaurus, being alternatively placed as the most basal archosauromorph to the exclusion of pterosaurs (Benton, 1985), as the sister-taxon of Rhynchosauria (Chatterjee, 1986), or as the sister-taxon of “Prolacertiformes” + Archosauriformes (Evans, 1988).

Figure 1 Phylogenetic trees depicting selected previous hypotheses for the higher-level relationships of early archosauromorphs in the period 1985–1996.

(A) Benton (1985); (B) Evans (1988); (C) Chatterjee (1986); (D) Gauthier, Kluge & Rowe (1988); and (E) Bennett (1996). Abbreviations: Ar., Archosauromorpha; Ara, Araeoscelidia; Arc., Archosauriformes; Arco., Archosauria; Di., Diapsida; Neo., Neodiapsida; Pr., Prolacertiformes; Sa., Sauria.

Gauthier (1994) recovered a monophyletic “Prolacertiformes” as the earliest branch of Archosauromorpha, whereas rhynchosaurs were the sister-taxon of Trilophosaurus and Archosauriformes, contrasting with previous analyses. Dilkes (1998), Sues (2003) and Modesto & Sues (2004) (the latter two analyses used a modified version of the data matrix of Dilkes (1998)) recovered the South African Prolacerta as more closely related to Archosauriformes than to any other archosauromorph, thus resulting in a polyphyletic “Prolacertiformes” (Fig. 2A). These authors found a monophyletic “Protorosauria” (formed by Protorosaurus, tanystropheids and drepanosaurs) as the earliest branching archosauromorphs and Trilophosaurus (together with Teraterpeton in the case of Sues et al. (2003)) as the sister-taxon of the clade composed of Rhynchosauria, Prolacerta and Archosauriformes. In addition, Dilkes (1998) and Gottmann-Quesada & Sander (2009) recovered the enigmatic choristoderan diapsids outside Sauria (i.e., Lepidosauromorpha and Archosauromorpha) (Fig. 2C), as also suggested by Evans & Hecht (1993).

Figure 2 Phylogenetic trees depicting selected previous hypotheses for the higher-level relationships of early archosauromorphs in the period 1998–2009.

(A) Dilkes (1998); (B) Müller (2004); and (C) Gottmann-Quesada & Sander (2009). Abbreviations: Ar., Archosauromorpha; Arc., Archosauriformes; Ch., Choristodera; Dr., Drepanosauridae; Rhy., Rhynchosauria; Rhyn., Rhynchosauridae; Sa., Sauria; Ta., Tanystropheidae.

Müller (2004) and Bickelmann, Müller & Reisz (2009) also recovered a non-monophyletic “Prolacertiformes,” in which Tanystropheus, Macrocnemus and Prolacerta were successive sister-taxa of a trichotomy composed of Trilophosaurus, Rhynchosauria and Archosauriformes (Fig. 2B). In these analyses, choristoderans were recovered as the earliest branching members of Lepidosauromorpha. Borsuk-Białynicka & Evans (2009) found results consistent with those of Dilkes (1998) and Müller (2004), respectively, based on slightly modified versions of those data matrixes. Gottmann-Quesada & Sander (2009) found Trilophosaurus as the sister-taxon of Rhynchosauria and “Protorosauria” as a paraphyletic group, with a clade composed of Protorosaurus and the drepanosaur Megalancosaurus found as the sister-taxon of tanystropheids and more crownward archosauromorphs. Gottmann-Quesada & Sander (2009) also found Prolacerta as the sister-taxon of Archosauriformes.

Ezcurra, Scheyer & Butler (2014) found a result largely consistent with that of Dilkes (1998), but noted that since the main purpose of their analysis was not to reconstruct the higher-level relationships of archosauromorphs, the recovery of a monophyletic “Protorosauria” was potentially an artefact of incomplete taxon and character sampling, and that Aenigmastropheus and Protorosaurus might in reality be more basal than tanystropheids (Fig. 3B). Pritchard et al. (2015) conducted a phylogenetic analysis that found a topology that clearly contrasts with those of the preceding 10 years. In this analysis, “Prolacertiformes” and “Protorosauria” were recovered as non-monophyletic groups, with Protorosaurus as the earliest branching archosauromorph and Prolacerta the sister-taxon of Archosauriformes (Fig. 3A). Pritchard et al. (2015) found rhynchosaurs, tanystropheids and a clade composed of Trilophosaurus spp. and Teraterpeton as successive sister-taxa of Prolacerta + Archosauriformes. Nesbitt et al. (2015) conducted another analysis based on a modified version of the data matrix of Pritchard et al. (2015), and found tanystropheids, rhynchosaurs, and allokotosaurians (a new group of archosauromorphs formed by Azendohsaurus, Trilophosaurus, and their kin) as successive sister-taxa of Prolacerta + Archosauriformes (Fig. 3C), thus being more consistent with previous analyses that repeatedly recovered tanystropheids as more basal than rhynchosaurs and crownward archosauromorphs (e.g., Gauthier, 1994; Dilkes, 1998; Müller, 2004; Gottmann-Quesada & Sander, 2009; Ezcurra, Scheyer & Butler, 2014).

Figure 3 Phylogenetic trees depicting selected previous hypotheses for the higher-level relationships of early archosauromorphs in the period 2014–2015.

(A) Pritchard et al. (2015); (B) Ezcurra, Scheyer & Butler (2014); and (C) Nesbitt et al. (2015). Abbreviations: All., Allokotosauria; Ar., Archosauromorpha; Ara, Araeoscelidia; Arc., Archosauriformes; Arco., Archosauria; Di., Diapsida; Lep., Lepidosauromorpha; Neo., Neodiapsida; Pro., Proterosuchidae; Prot., Protorosauria; Proto., Protorosauridae; Rh., Rhynchocephalia; Rhy., Rhynchosauria; Sa., Sauria; Saur., Sauropsida; Ta., Tanystropheidae; Young., Younginiformes.

Cladistic analyses focused on the higher-level phylogenetic relationships of Archosauromorpha have found rather disparate results over the last 30 years, but, in the last decade, most analyses have agreed on the polyphyletic nature of “Prolacertiformes” as traditionally conceived and the placement of protorosaurs/tanystropheids as the sister-taxon of rhynchosaurs, Prolacerta, and archosauriforms. However, there is still a substantial lack of consensus as to the monophyly and taxonomic content of “Protorosauria” and the position of allokotosaurians also represents a topic to be explored in the coming years.

Figure 4 Chronostratigraphic diagram of proterosuchian-bearing units (sensu Ezcurra, Butler & Gower, 2013; Ezcurra et al., 2015).

Ages of South African units based on Rubidge (2005) and Rubidge et al. (2013); Russian units based on Newell et al. (2010) and Newell et al. (2012); Chinese and Indian units based on Lucas (2010) and Liu, Li & Li (2013); South American units based on Piñeiro et al. (2003) and Spalletti, Fanning & Rapela (2008); and Australian units based on Ezcurra (2014) and Warren & Hutchinson (1990). Asterisks indicate radioistopically dated boundaries. It should be noted that the lower Ermaying and Heshanggou formations of China, and the South American, Indian and Australian units belong to different basins, respectively. Russian Gorizonts (=Horizonts) include several formations and basins. Abbreviations: Ans, Anisian; AZ, Assemblage Zone; Chx, Changhsingian; Crn, Carnian; Fm, Formation; G, Gorizont; Ind, Induan; Lad, Ladinian; Nor, Norian; Ole, Olenekian; Ryb, Rybinskian; Slu, Sludkian; Sub, Subzone; Supergo, Supergorizont; Ust, Ustmylian; Vok, Vokhmian; Wuc, Wuchiapingian. Geological timescale after Gradstein et al. (2012).

Historical background of the phylogenetic relationships of “Proterosuchia”

The first discovered proterosuchian fossil was collected in the Lower Triassic Panchet Formation of India (Fig. 4) and described by Huxley (1865) as a new genus and species “Ankistrodon indicus.” This species is based on a fragment of tooth-bearing bone interpreted by Huxley (1865: 12) as a “thecodont saurian” with a tooth morphology closely resembling that of other carnivorous “thecodonts” and dinosaurs. During the early 20th century, Broom (1903a) described the remains of a fossil reptile collected in the Lower Triassic part of the Karoo Basin of Eastern Cape Province, South Africa, and erected the new species Proterosuchus fergusi. He stated that Proterosuchus differed so greatly from any hitherto described species that it was difficult to decide its affinities (Broom, 1903a: 162). However, mainly based on the morphology of the palatal teeth, Broom (1903a: 163) concluded that Proterosuchus was a primitive “Rhynchocephalian” (conceived by Broom as a group of primitive reptiles including the extant Sphenodon and taxa like the extinct Procolophon and Protorosaurus) that showed “a considerable degree of specialisation along the line which gave rise to the early Crocodiles and Dinosaurs.” Two years later, the same author named Erythrosuchus africanus from the early Middle Triassic of South Africa and assigned it to the Phytosauria (Broom, 1905). Subsequently, Broom (1906) reviewed the classification of “Diaptosauria” (a group that included several amniote clades, such as protorosaurs, pelycosaurs, rhynchosaurs, procolophonids, choristoderans and rhynchocephalians; sensu Osborn, 1903) and coined the new suborder or order “Proterosuchia,” within which he included Proterosuchus. Broom (1906) interpreted the “Proterosuchia” as more closely related to the Rhynchocephalia than to other Triassic diaptosaurian orders, such as Phytosauria and “Gnathodontia” (e.g., Howesia).

Huene (1908) considered “Proterosuchia” as a family-ranked group, resulting in the new taxon Proterosuchidae. Subsequently, Huene (1908) described a new and more complete specimen of Erythrosuchus and proposed the new order “Pelycosimia,” in which he included Erythrosuchus and the derived rhynchosaur “Scaphonyx” (=Hyperodapedon sensu Langer et al., 2000), among other taxa (Huene, 1911). Huene (1911) interpreted the “Pelycosimia” as closely related to pelycosaurian synapsids. However, Watson (1917) recognized Erythrosuchus as a member of the “Thecodontia” and coined the monospecific, family-ranked clade Erythrosuchidae. Subsequently, Huene (1920) agreed with this new interpretation and included the “Pelycosimia” within the order “Thecodontia” as a suborder.

Haughton (1924) described the new species “Chasmatosaurus vanhoepeni” from the same horizon as Proterosuchus fergusi, and assigned it to its own family, Chasmatosauridae. Willistons (1925) considered the Proterosuchia as a non-thecodont order of diapsids. Subsequently, the suborder Erythrosuchia was erected by Goodrich (1930) in order to include only Erythrosuchus. Kuhn (1933) considered Proterosuchus to be closely related to the probable stem-crocodylomorphs Dyoplax and Erpetosuchus, whereas he interpreted Erythrosuchus and “Chasmatosaurus” as forming a group of closely related taxa together with the aetosaur “Acompsosaurus.” Broili & Schröder (1934) also considered “Chasmatosaurus” and Erythrosuchus to be closely related.

Between the 1940s and 1960s several classifications were proposed that mostly agreed in considering proterosuchids and erythrosuchids as closely related taxa forming a single group (“Proterosuchia” or Proterosuchoidea, depending on the rank employed by each author) within a primitive stock of thecodonts (e.g., Romer, 1945; Romer, 1956; Huene, 1948; Huene, 1956; Hoffstetter, 1955; Kuhn, 1961; Reig, 1961). Ochev (1958) described Garjainia prima from the late Early Triassic of Russia and included it in its own family, Garjainiidae. Similarly, Huene (1960) described “Vjushkovia” triplicostata from the same horizon as Garjainia prima and erected the new family Vjushkoviidae, and Young (1964) described Shansisuchus shansisuchus from the Middle Triassic of China and coined the family Shansisuchidae. These families were included within the “Proterosuchia,” together with Proterosuchidae and Erythrosuchidae, by some authors (e.g., Young, 1964; Reig, 1961; Kuhn, 1966). Tatarinov (1961) proposed that “Vjushkovia”, Garjainia, “Dongusia,” and Cuyosuchus were subjective junior synonyms of Erythrosuchus, a hypothesis that was followed by several other authors (e.g., Hughes, 1963; Ewer, 1965; Romer, 1966; Romer, 1972a; Cruickshank, 1972), but rejected by Young (1964),Charig & Reig (1970), and subsequent workers (e.g., Parrish, 1992; Sennikov, 1995a; Sennikov, 1995b; Gower & Sennikov, 1996; Gower & Sennikov, 1997; Gower & Sennikov, 2000; Desojo, Arcucci & Marsicano, 2002; Ezcurra, Lecuona & Martinelli, 2010; Ezcurra, Butler & Gower, 2013; Ezcurra, Scheyer & Butler, 2014; Ezcurra & Butler, 2015a; Ezcurra & Butler, 2015b).

Charig & Reig (1970) recognized the suborder Proterosuchia as composed of two families: Proterosuchidae (“Chasmatosaurus”-like forms) and Erythrosuchidae (Erythrosuchus-like forms). These authors included Archosaurus, “Chasmatosaurus”, Chasmatosuchus, “Elaphrosuchus,” and Proterosuchus within the Proterosuchidae, and Garjainia, Erythrosuchus, “Vjushkovia,” Shansisuchus, and possibly Cuyosuchus within the Erythrosuchidae. Accordingly, Charig & Reig (1970) considered Shansisuchidae, Garjainiidae and Vjushkoviidae as junior synonyms of Erythrosuchidae, and Chasmatosauridae as a junior synonym of Proterosuchidae. Charig & Reig (1970) considered the proterosuchids as a primitive stock of “thecodonts” from which erythrosuchids evolved. In particular, “Elaphrosuchus” was depicted as more closely related to erythrosuchids than were other proterosuchids (Charig & Reig, 1970: Fig. 6). Subsequently, Bonaparte (1982) proposed a more inclusive suborder “Proterosuchia,” being composed of two distinct infraorders: Proterochampsia and “Rauisuchia.” Within Proterochampsia, Bonaparte (1982) included Proterosuchidae, Cerritosauridae and Proterochampsidae, whereas “Rauisuchia” was composed of Erythrosuchidae and Rauisuchidae. From the late 1970s to early 1990s, multiple Early and Middle Triassic archosauriforms were described from Australia, Russia and China (Camp & Banks, 1978; Ochev, 1979; Ochev, 1980; Thulborn, 1979; Cheng, 1980; Wu, 1981; Peng, 1991; Sennikov, 1992; Sennikov, 1994), and most were assigned to either Proterosuchidae or Erythrosuchidae following the “Chasmatosaurus”-like and Erythrosuchus-like dichotomy.

Benton (1985) reported the first cladistic analysis that included early archosauriforms, although it still was not a numerical (fully explicit) approach. Benton (1985) placed Proterosuchus fergusi either within “Prolacertiformes” or as the sister-taxon of his concept of Archosauria (which in Benton’s usage was a non-crown-group clade including Erythrosuchus, Euparkeria, and “later archosaurs”), with Erythrosuchus africanus recovered as the sister-taxon to all other archosaurs. The first numerical cladistic analysis to include proterosuchians was that of Gauthier (1986), who employed Proterosuchidae to root the tree in which he found Erythrosuchidae as the sister-taxon of Proterochampsidae + Archosauria. Gauthier (1986: 42) restricted the usage of the term Archosauria to the crown group, and Gauthier, Kluge & Rowe (1988) erected the new group Archosauriformes for the clade including all the descendants of the most recent common ancestor of Proterosuchidae, Erythrosuchidae, Proterochampsidae and Archosauria. Several subsequent analyses recovered broadly similar topologies to that of Gauthier (1986), including those of Benton & Clark (1988), Sereno & Arcucci (1990), Sereno (1991), Parrish (1993), Gower (2002) and Benton (2004), but the paraphyly of “Proterosuchia” in these studies was the outcome of choices (implicit or explicit) in rooting and not a result of the analyses (other than that erythrosuchids lie outside of non-proterosuchid, non-erythrosuchid archosauriforms when trees are rooted with proterosuchids). Juul (1994) and Bennett (1996) were the first authors to include both Proterosuchidae and Erythrosuchidae as part of the ingroup in numerical analyses, and, as a consequence, to test the phylogenetic position of proterosuchids among archosauromorphs. The analyses of Juul (1994) and Bennett (1996) recovered proterosuchids and erythrosuchids as successive outgroups of all other archosauriforms, though proterosuchids and erythrosuchids were each represented either as a suprageneric taxon or by a single species, such that the monophyly of Proterosuchidae and Erythrosuchidae were not tested. Parrish (1992) reported an analysis that included six proterosuchian species and an aggregate “other archosauriforms” in the ingroup and Proterosuchus as an outgroup. Within his monophyletic Erythrosuchidae, Shansisuchus shansisuchus, Erythrosuchus africanus, Garjainia prima and Fugusuchus hejiapensis were successive outgroups of a monophyletic “Vjushkovia,” comprising “Vjushkovia” triplicostata (=Garjainia triplicostata) and “Vjushkovia” sinensis (=Youngosuchus sinensis) (Fig. 5A).

Figure 5 Phylogenetic trees depicting selected previous hypotheses of relationships for “Proterosuchia.”

(A) Single most parsimonious tree of Parrish (1992); (B) strict reduced consensus tree of Gower & Sennikov (1997); (C) single most parsimonious tree of Dilkes & Sues (2009); (D) strict consensus tree of Ezcurra, Lecuona & Martinelli (2010); and (E) strict consensus tree of Nesbitt (2011). Bremer support (= Decay Index) values greater than one are indicated below each node (support values for the trees of Parrish (1992) and Gower & Sennikov (1997) were calculated using the original data matrices). Abbreviations: Erythr., Erythrosuchidae; Protero., Proterosuchidae. Bold font indicates putative proterosuchians.

Gower & Sennikov (1996) reported a cladistic analysis based only on braincase characters, in which proterosuchians were recovered as monophyletic. Among proterosuchians, distinct proterosuchid and erythrosuchid clades were recognized, with Fugusuchus hejiapensis as well as Proterosuchus fergusi found as members of Proterosuchidae (contra Parrish, 1992). Xilousuchus sapingensis, “Vjushkovia” triplicostata, Erythrosuchus africanus, and Shansisuchus shansisuchus were recovered as members of Erythrosuchidae (Gower & Sennikov, 1996). Within Erythrosuchidae, Gower & Sennikov (1996) found sister-taxon relationships for Xilousuchus sapingensis and “Vjushkovia” triplicostata, and for Erythrosuchus africanus and Shansisuchus shansisuchus. Gower & Sennikov (1997) added non-braincase characters and included a different taxon sampling and recovered a paraphyletic “Proterosuchia,” with erythrosuchids more closely related to Archosauria than to proterosuchids (Fig. 4B). Proterosuchus fergusi was found as sister-taxon to the other putative proterosuchids Fugusuchus hejiapensis and Sarmatosuchus otschevi. Additionally, in contrast to the earlier study of Gower & Sennikov (1996), Xilousuchus sapingensis was recovered as an erythrosuchid in only some of their most parsimonious trees (Gower & Sennikov, 1997). Gower & Sennikov (2000) conducted a review of the Permo–Triassic archosauriforms from Russia and concluded that “Vjushkovia” was a junior synonym of Garjainia, in agreement with previous comments by Kalandadze & Sennikov (1985), Ochev & Shishkin (1988) and Sennikov (1995a) and Sennikov (1995b), and that Garjainia triplicostata was probably also a junior synonym of Garjainia prima.

Recent work by Dilkes & Sues (2009) found results that differ from most previous phylogenetic analyses. Although Proterosuchus fergusi was recovered as the sister-taxon of all other archosauriforms, Euparkeria capensis was found as the sister-taxon of Erythrosuchus africanus and more crownward archosauriforms (Fig. 5C). Thus, Dilkes & Sues (2009) recovered a polyphyletic (rather than the more usual paraphyletic) “Proterosuchia.” Subsequent revision of this dataset by Dilkes & Arcucci (2012) recovered a paraphyletic “Proterosuchia.” Ezcurra, Lecuona & Martinelli (2010) described the new early archosauriform Koilamasuchus gonzalezdiazi and expanded the taxonomic and character sample of the data matrix of Dilkes & Sues (2009), including characters employed in several previous archosauriform phylogenetic analyses (e.g., Dilkes, 1998; Gower & Sennikov, 1997). Among taxa sampled by Ezcurra, Lecuona & Martinelli (2010), the taxonomic content of Proterosuchidae hypothesized by previous authors (e.g., Proterosuchus, Sarmatosuchus and Fugusuchus: Gower & Sennikov, 1997) was recovered as paraphyletic (or even polyphyletic because the specimen that it is now the holotype of Koilamasuchus gonzalezdiazi was interpreted originally as a proterosuchid: Bonaparte, 1981), and the previously hypothesized content of Erythrosuchidae (Erythrosuchus africanus, Shansisuchus shansisuchus, Garjainia triplicostata) was recovered as monophyletic (Fig. 5D). Ezcurra, Lecuona & Martinelli (2010) recovered Proterosuchus fergusi as sister-taxon to all other sampled archosauriforms, with Sarmatosuchus otschevi and Fugusuchus hejiapensis (considered as proterosuchids by Gower & Sennikov, 1996; Gower & Sennikov, 1997; Gower & Sennikov, 2000) being more closely related to other archosauriforms than to Proterosuchus fergusi. Ezcurra, Lecuona & Martinelli (2010) also recovered Koilamasuchus gonzalezdiazi and the proposed euparkeriid Osmolskina czatkowicensis (Borsuk-Białynicka & Evans, 2003) as successive outgroups of Erythrosuchidae and more crownward archosauriforms. Erythrosuchidae was recovered as the sister-group of Euparkeria capensis and more crownward archosauriforms, in contrast to the results of Dilkes & Sues (2009) but in agreement with most other quantitative phylogenetic analyses of Archosauriformes (e.g., Juul, 1994; Bennett, 1996; Gower & Sennikov, 1997; Nesbitt et al., 2009; Nesbitt, 2011). Within Erythrosuchidae, Ezcurra, Lecuona & Martinelli (2010) recovered Shansisuchus shansisuchus as the sister-taxon of Erythrosuchus africanus + “Vjushkovia” (=Garjainia) triplicostata, contrasting with the internal relationships of Erythrosuchidae found by Gower & Sennikov (1996). Desojo, Ezcurra & Schultz (2011) obtained broadly similar results to those of Ezcurra, Lecuona & Martinelli (2010) using a similar data matrix.

Nesbitt (2011) recovered Erythrosuchus africanus and Proterosuchidae as successive outgroups of all non-proterosuchian archosauriforms, resembling most previous numerical analyses. Nesbitt (2011: 249) was the first to include the oldest known archosauriform, Archosaurus rossicus, in a numerical phylogenetic analysis and found it to be the sister-taxon of Proterosuchus fergusi within a monophyletic Proterosuchidae (Fig. 5E). Nesbitt (2011) and Nesbitt, Liu & Li (2011) reinterpreted the Chinese taxon Xilousuchus sapingensis, once considered a member of “Proterosuchia” (see above), as a poposauroid archosaur, a hypothesis followed by recent authors (e.g., Butler et al., 2011). Ezcurra, Scheyer & Butler (2014) revised the Permian saurian record and conducted a phylogenetic analysis including several basal archosauromorphs. These authors found Proterosuchus fergusi and Archosaurus rossicus within a monophyletic Proterosuchidae, in agreement with Nesbitt (2011) (Fig. 3B). The enigmatic Permian diapsid Eorasaurus olsoni was recovered in a polytomy together with Erythrosuchus africanus and Euparkeria capensis, thus potentially representing the oldest known archosauriform (Ezcurra, Scheyer & Butler, 2014). More recently, Nesbitt et al. (2015) included the Chinese archosauriform “Chasmatosaurus” yuani for the first time in a quantitative phylogenetic analysis and recovered it within Proterosuchidae, being the sister-taxon of Proterosuchus spp. (Fig. 3C).

In summary, the systematic history of proterosuchians was chaotic through most of the 20th century, but the advent of numerical phylogenetic techniques gave rise to near-unanimous consensus regarding the non-monophyly of “Proterosuchia” (but see Gower & Sennikov, 1996), with erythrosuchids being more closely related to Archosauria than to proterosuchids. However, this stability may be superficial because few proterosuchian species have been included in numerical phylogenetic analyses. Furthermore, the taxonomic contents and internal relationships of Proterosuchidae and Erythrosuchidae have not yet been tested thoroughly. It is important to note that quantitative support for many of the previously recovered relationships among putative proterosuchids and erythrosuchids is low in terms of decay indices, and that topology-based statistical tests of the (non)monophyly of Proterosuchidae, Erythrosuchidae and Proterosuchia have not been carried out. The phylogenetic analysis presented here is intended to shed light on these issues.

Materials and Methods

Objectives and taxonomic sample

The aim of the present phylogenetic analysis is to generate a comprehensive higher-level phylogenetic hypothesis of basal archosauromorphs and shed light on the species-level interrelationships of taxa historically identified as proterosuchian archosauriforms (i.e., taxa usually considered as members either of Proterosuchidae or Erythrosuchidae; Charig & Reig, 1970; Charig & Sues, 1976; Ezcurra, Butler & Gower, 2013). As a result, the taxonomic sample is mainly focused on non-archosaurian archosauromorphs and, more specifically, on proterosuchians, which range chronostratigraphically from the upper Permian (Archosaurus rossicus) to the Upper Triassic (Cuyosuchus huenei). Six non-archosauromorph diapsids were included as outgroups: the early diapsid Petrolacosaurus kansensis, the basal neodiapsids Youngina capensis and Acerosodontosaurus piveteaui, and the early lepidosauromorphs Paliguana whitei, Planocephalosaurus robinsonae and Gephyrosaurus bridensis. All of these taxa have been consistently recovered outside Archosauromorpha in recent phylogenetic analyses (Müller, 2004; Bickelmann, Müller & Reisz, 2009; Reisz, Laurin & Marjanović, 2010; Ezcurra, Scheyer & Butler, 2014), and, as a whole, provide an exhaustive sample of early diapsid character states. The late Carboniferous Petrolacosaurus kansensis has been repeatedly found to be more distantly related to archosauromorphs than are Youngina capensis, Acerosodontosaurus piveteaui, and lepidosauromorphs (Müller, 2004; Senter, 2004; Bickelmann, Müller & Reisz, 2009; Reisz, Modesto & Scott, 2011; Ezcurra, Scheyer & Butler, 2014) and therefore was chosen here to root the phylogenetic trees.

The taxonomic sample of non-archosauriform archosauromorphs is chosen in order to test the higher-level phylogenetic relationships between relatively well-established groups (e.g., Rhynchosauria, Tanystropheidae) and several species with problematic affinities, such as several taxa usually assigned to the likely non-monophyletic “Prolacertiformes.” A total of 19 taxa previously identified as non-archosauriform archosauromorphs are included, including three tanystropheids (Macrocnemus bassanii, Amotosaurus rotfeldensis, Tanystropheus longobardicus), six rhynchosaurs (Noteosuchus colletti, Mesosuchus browni, Howesia browni, Eohyosaurus wolvaardti, Rhynchosaurus articeps, Bentonyx sidensis), eight taxa identified previously as protorosaurs/prolacertiforms (Aenigmastropheus parringtoni, Protorosaurus speneri, Prolacertoides jimusarensis, Boreopricea funerea, Jesairosaurus lehmani, Prolacerta broomi, Kadimakara australiensis, Eorasaurus olsoni), and three allokotosaurians (Pamelaria dolichotrachela, Trilophosaurus buettneri, Azendohsaurus madagaskarensis). Non-proterosuchian archosauriforms are represented by 38 taxa, including four doswelliids (Archeopelta arborensis, Tarjadia ruthae, Jaxtasuchus salomoni, Doswellia kaltenbachi), all known proterochampsid species (see Trotteyn, Arcucci & Raugust, 2013), four basal phytosaurs (Parasuchus agustifrons, Parasuchus hislopi, Nicrosaurus kapffi, Smilosuchus spp.), seven basal ornithodirans (Dimorphodon macronyx, Lagerpeton chanarensis, Marasuchus lilloensis, Lewisuchus admixtus, Silesaurus opolensis, Heterodontosaurus tucki, Herrerasaurus ischigualastensis), two ornithosuchids (Ornithosuchus longidens, Riojasuchus tenuisceps), seven basal suchians (Aetosauroides scagliai, Gracilisuchus stipanicicorum, Turfanosuchus dabanensis, Prestosuchus chiniquensis, Batrachotomus kupferzellensis, Nundasuchus songeaensis, Yarasuchus deccanensis), and five archosauriforms that seem not to fit into any of the aforementioned clades (Euparkeria capensis, Asperoris mnyama, Dorosuchus neoetus, Dongusuchus efremovi, Vancleavea campi). Smilosuchus spp. is scored as a supraspecific terminal because the taxonomy of this phytosaur taxon is currently problematic, although it appears to represent a monophyletic genus (Stocker, 2010; Stocker & Butler, 2013).

The preferred option here was to sample suprageneric taxa using multiple species-level terminals rather than a composite suprageneric terminal because it has been shown by previous authors that the former method considerably outperforms the latter in both simulations and empirical data (Wiens, 1998; Prendini, 2001; Jenner, 2006; Brusatte, 2010). As a result, basal representatives of each clade were sampled (e.g., Noteosuchus colletti, Mesosuchus browni, Howesia browni and Eohyosaurus wolvaardti for rhynchosaurs; Lagerpeton chanarensis, Marasuchus lilloensis, and Lewisuchus admixtus for ornithodirans; Parasuchus angustifrons and Parasuchus hislopi for phytosaurs) as well as species that represent a balance between completeness (thus maximising phylogenetic data) and a relatively basal position in the lineage (e.g., Rhynchosaurus articeps and Bentonyx sidensis for rhynchosaurs; Silesaurus opolensis, Heterodontosaurus tucki and Herrerasaurus ischigualastensis for ornithodirans; Aetosauroides scagliai, Gracilisuchus stipanicicorum, Turfanosuchus dabanensis, Prestosuchus chiniquensis, Batrachotomus kupferzellensis and Nundasuchus songeaensis for suchians).

The proterosuchian sample is intended to be the most comprehensive of the data set, representing a total of 32 terminals that sample all currently valid nominal species (Ezcurra, Butler & Gower, 2013). The holotype of Proterosuchus fergusi (SAM-PK-591) was considered undiagnostic by Ezcurra & Butler (2015a), but this specimen is included as an independent terminal in order to test its phylogenetic position. It is noteworthy that a number of proterosuchian species are included here for the first time in a quantitative phylogenetic analysis, namely “Blomosuchus georgii,” Vonhuenia friedrichi, “Ankistrodon indicus,” Proterosuchus alexanderi, Proterosuchus goweri, Kalisuchus rewanensis, Chasmatosuchus rossicus, “Gamosaurus lozovskii,” “Exilisuchus tubercularis,” Uralosaurus magnus, Shansisuchus kuyeheensis, Cuyosuchus huenei, Guchengosuchus shiguaiensis, Chalishevia cothurnata and Garjainia madiba. Also included are an unnamed taxon represented by isolated dorsal vertebrae from the Early Triassic of southeastern Australia (Kear, 2009), a partial skeleton of a small erythrosuchid from the Cynognathus Assemblage Zone (AZ) of South Africa that was previously considered possibly referable to Erythrosuchus africanus (Gower, 2003), and the probable pseudosuchian “Dongusia colorata” (Gower & Sennikov, 2000). Several species included in the taxonomic sample of the analysis are currently considered as nomina dubia (e.g., “Ankistrodon indicus,” “Gamosaurus lozovskii,” “Dongusia colorata”; Charig & Reig, 1970; Charig & Sues, 1976; Gower & Sennikov, 2000; Ezcurra, Butler & Gower, 2013), but they were included in this data matrix to test their original phylogenetic interpretation and the taxonomic content of Proterosuchidae and Erythrosuchidae. “Crenelosaurus nigrosilvanus,” “Ocolurtaia arquata,” “Seemannia palaeotriadica,” and “Shansisuchus heiyuekouensis” represent four poorly informative proterosuchian species considered nomina dubia (Gower & Sennikov, 2000; Ezcurra, Butler & Gower, 2013) and, as a result, are not included in the analysis. A specifically indeterminate occurrence of “Chasmatosaurus” from the earliest Triassic of India (Satsangi, 1964) is also based on very fragmentary bones and is not also included.

The assignment of the referred specimens of Kadimakara australiensis, Garjainia madiba and Uralosaurus magnus is problematic (see below Remarks about these species) and, as a result, the holotype and the complete hypodigm (i.e., type and referred specimens) of these species were scored as independent terminals. The aim of these independent scorings is to test the effect that the referred material of the hypodigm has on the phylogenetic relationships of the species. Following a similar logic, Chasmatosuchus magnus (=“Jaikosuchus magnus”) and “Gamosaurus lozovskii” were scored as independent terminals in a first analysis and were merged into a single terminal in a second analysis because they are considered here subjective synonyms between each other (see below Remarks about these species). These alternative scorings allow the hypothesis of synonymy to be tested.

The first-hand study of Archosaurus rossicus, “Blomosuchus georgii,” Vonhuenia friedrichi, Chasmatosuchus magnus (=“Jaikosuchus magnus”), “Gamosaurus lozovskii,” Dorosuchus neoetus, Dongusuchus efremovi and Kalisuchus rewanensis as part of this research resulted in modifications to their taxonomy and/or casted doubts about the composition of their hypodigms (see Remarks about these species below). The specimens that have been used for the scorings of these species are summarized as follows: (i) Archosaurus rossicus and (ii) Kalisuchus rewanensis are scored based only on their holotypes; (iii) “Blomosuchus georgii” is scored based on its holotype and an isolated parabasisphenoid previously referred to Vonhuenia friedrichi (PIN 1025/14); (iv) Vonhuenia friedrichi is scored based on its holotype and a referred cervical vertebra (PIN 1025/419); (v) Chasmatosuchus magnus is scored based on its holotype; (vi) “Gamosaurus lozovskii” is scored based on its holotype and five referred vertebrae (PIN 3361/14, 94, 183, 213, 214); (vii) Dorosuchus neoetus is scored based on its type series; and (viii) Dongusuchus efremovi is scored based on its holotype and four unambiguously referred isolated femora.

The resultant taxonomic list of the data matrix is composed of 96 independent taxa, plus the holotype of Proterosuchus fergusi, three hypodigms that partially overlap with the scorings of their respective holotypes (Kadimakara australiensis, Garjainia madiba, and Uralosaurus magnus) and two species that are probably synonyms (“Gamosaurus lozovskii” and Chasmatosuchus magnus) (i.e., 102 operational taxonomic units). The vast majority of species included in the archosauromorph taxonomic sample are Triassic in age, with the exception of four late Permian (Aenigmastropheus parringtoni, Archosaurus rossicus, Eorasaurus olsoni, and Protorosaurus speneri) and three Early Jurassic (Gephyrosaurus bridensis, Dimorphodon macronyx, and Heterodontosaurus tucki) species.

Operational taxonomic units

The terminals used in this phylogenetic analysis are listed as follows.

Petrolacosaurus kansensis Lane, 1945 (OUTGROUP)

Age. Kasimovian–Gzhelian, Late Pennsylvanian, latest Carboniferous (Reisz, 1977; Reisz & Berman, 1986).

Locality. Garnett Quarry, Missourian Series, Anderson County, Kansas, midwest USA (Reisz, 1977; Reisz & Berman, 1986).

Stratigraphic horizon. Rock Lake Member, Stanton Formation, Lansing Group (Reisz, 1977; Reisz & Berman, 1986).

Holotype. KUVP 1424: partial right hindlimb of an adult individual.

Referred material. Multiple immature and mature specimens housed in KUPV and listed by Reisz (1981: 4, 5).

Diagnosis. Early diapsid distinguished from other diapsids on the basis of the following unique combination of characters: well developed supratemporal and infratemporal fenestrae and elongate; narrow suborbital fenestra; parietal without posterolateral process; posterior splenial bone present; marginal dentition unusually thin-walled; six elongate cervical vertebrae; a pair of mammillary processes on the neural spine of first sacral vertebra; large dorsal ischiadic notch; slender forelimbs equal in length to more massive hindlimb; and propodials equal in length to epipodials (Reisz, 1981: 4).

Remarks. Petrolacosaurus kansensis is one of the oldest and more completely known early diapsid (Reisz, 1977) and has been usually included as one of the outgroups in phylogenetic analyses focused on the internal relationships of neodiapsids or basal saurians (e.g., Benton, 1985; Jalil, 1997; Dilkes, 1998; Modesto & Reisz, 2002; Senter, 2004; Modesto & Sues, 2004; Pritchard et al., 2015). Comprehensive osteological descriptions of this species were published by Peabody (1952) and Reisz (1981).

Acerosodontosaurus piveteaui Currie, 1980

Age. Wuchiapingian–Changhsingian, late Permian (Currie, 1980).

Locality. Sakamena River Valley, Tulear, southern Madagascar (Currie, 1980).

Stratigraphic horizon. Lower Sakamena Formation, Sakamena Group (Currie, 1980).

Holotype. MNHN 1908-32-57: partial skeleton of an immature individual mostly preserved as natural moulds in a single concretion preserving part and counterpart (Currie, 1980). The skeleton lacks the anterior portion of the snout, left side of the skull, braincase, tail, most of the pectoral girdle, left manus, left hindlimb, and lower part of the right hindlimb.

Emended diagnosis. Early neodiapsid distinguished from other diapsids on the basis of the following unique combination of character-states: ventrally open infratemporal fenestra; quadratojugal absent; at least 36 tooth positions in the maxilla and 34 in the dentary; cleithrum present; radius twisted; ulna lacking an ossified olecranon; and pubis with long tubercle (modified from Bickelmann, Müller & Reisz, 2009).

Remarks. Acerosodontosaurus piveteaui was originally described by Currie (1980) and more recently redescribed by Bickelmann, Müller & Reisz (2009). The only known specimen consists of natural, well-preserved moulds that provide a very good anatomical record of the postcranial axial skeleton of early neodiapsids.

Youngina capensis Broom, 1914

Age. Capitanian-Changhsingian, latest middle to late Permian (Rubidge et al., 2013).

Localities. New Bethesda, Graaff-Reinet, Eastern Cape Province, South Africa (type locality; Gow, 1975; Smith & Evans, 1996; Gardner, Holliday & O’Keefe, 2010).

Stratigraphic horizons. Dicynodon AZ, Balfour Formation, Karoo Supergroup, and referred specimens have been also collected in the Tropidostoma AZ of the Middelton and Balfour formations, Karoo Supergroup, South Africa (Gow, 1975; Smith & Evans, 1996; Gardner, Holliday & O’Keefe, 2010).

Holotype. AMNH 5561: skull and vertebrae.

Referred material. Multiple specimens housed in the United States and South Africa, and listed by Gow (1975: 90). Subsequently, the following specimens were referred to Youngina capensis, SAM-PK-K10818: cluster of disarticulated juvenile skeletons (Smith & Botha-Brink, 2014); SAM-PK-K10777: anterior skull and lower jaws with some postcranial bones (Smith & Botha-Brink, 2014); SAM-PK-K6205: partial skull; SAM-PK-K7578: dorsoventrally compressed skull; SAM-PK-K7710: an aggregation of five fairly complete to more fragmentary skeletons belonging to immature individuals (Smith & Evans, 1996); SAM-PK-K8565: partial skeleton; GHG K106: dorsoventrally compressed skull and lower jaws; and GHG RS160: partial skull.

Emended diagnosis. Early neodiapsid distinguished from other diapsids on the basis of the following unique combination of character-states: quadratojugal closing the infratemporal fenestra ventrally; long posterolateral process of the frontal; a pair of tabular bones; robust stapes bearing a stapedial foramen; parabasisphenoid with thick parasphenoid crests; radius twisted; ulna lacking an ossified olecranon; pelvic girdle lacking a thyroid fenestra; and metatarsal V with an outer process.

Remarks. Multiple juvenile to adult specimens of Youngina capensis have been collected in the last century and this species represents the most comprehensively studied Permian neodiapsid (Broom, 1914; Broom, 1921; Broom, 1922; Gow, 1975; Currie, 1981; Evans, 1987; Smith & Evans, 1996; Gardner, Holliday & O’Keefe, 2010). Youngina capensis has been repeatedly used in quantitative phylogenetic analyses as a key terminal to reconstruct the ancestral character-states of Sauria (e.g., Benton & Allen, 1997; Dilkes, 1998; Senter, 2004; Gottmann-Quesada & Sander, 2009; Ezcurra, Scheyer & Butler, 2014; Pritchard et al., 2015).

Paliguana whitei Broom, 1903b

Age. Induan–?early Olenekian, Early Triassic (Damiani et al., 2000; Rubidge, 2005; Lucas, 2010).

Locality. Donnybrook, between Tarkastad and Queenstown, South Africa (Carroll, 1975; Kitching, 1977; Groenewald & Kitching, 1995).

Stratigraphic horizon. Lystrosaurus AZ, upper Balfour Formation or lower Katberg Formation, Beaufort Group, Karoo Supergroup (Carroll, 1975; Kitching, 1977; Groenewald & Kitching, 1995).

Holotype. AM 3585: partial skull and lower jaw.

Emended diagnosis. Early saurian distinguished from other diapsids on the basis of the following unique combination of characters-states: at least some maxillary crowns with a convex distal margin; elongated and continuously dorsally curved anterior process of the jugal; anteroventrally oriented ventral process of the squamosal; well-exposed supratemporal fossa medial to the supratemporal fenestra on the parietal; and a quadrate conch.

Remarks. Paliguana whitei historically has been considered as the oldest known lepidosauromorph because of the presence of a quadrate conch (Evans & Borsuk-Białynicka, 2009; Evans & Jones, 2010). This species was recently included for the first time in a quantitative phylogenetic analysis and its position within Lepidosauromorpha was supported (Ezcurra, Scheyer & Butler, 2014). The holotype and only known specimen of Paliguana whitei has been recently further prepared (W De Klerk, pers. comm., 2012) and new information from the palatal region of the skull is available. Unfortunately, the poor state of preservation of these regions hampers the determination of several anatomical features (AM 3585).

Planocephalosaurus robinsonae Fraser, 1982

Age. Rhaetian, Late Triassic (Whiteside & Marshall, 2008).

Locality. Cromhall quarry, south Gloucestershire, UK (Fraser, 1982).

Stratigraphic horizon. Karstic fissures in Dinantian Limestones (Fraser, 1982).

Holotype. AUP No. 11061: left maxilla.

Referred material. Approximately 750 bones housed in the AUP (see Fraser, 1982; Fraser & Walkden, 1984) and the NHMUK PV.

Diagnosis. Early rhynchocephalian distinguished from other diapsids on the basis of the following unique combination of character-states: incomplete lower temporal bar; frontals and parietals fused; broad and flat parietal table with a large central pineal foramen; no supratemporal or lacrimal; deep overlap of the pterygoid and quadrate; quadrate and quadratojugal fused; quadrate foramen present; premaxilla paired; vomers with small scattered teeth; pterygoid with two tooth rows; no teeth on the transverse ramus of the pterygoid; palatines with two rows of conical teeth parallel to the marginal dentition; dentition acrodont; all teeth are radially ribbed; dentary with a posterior process that articulates with the articular complex; and no splenial (Fraser, 1982).

Remarks. Planocephalosaurus robinsonae is one of the best-known early rhynchocephalian lepidosauromorphs and possesses an intermediate morphology between more basal saurians and more advanced members of Rhynchocephalia, such as the species of Clevosaurus (Fraser, 1982; Evans & Jones, 2010). This species has been used in previous higher-level phylogenetic analyses as a representative of the early morphological diversity of Lepidosauromorpha (e.g., Reisz, Laurin & Marjanović, 2010). Planocephalosaurus robinsonae was described in detail by Fraser (1982) and Fraser & Walkden (1984).

Gephyrosaurus bridensis Evans, 1980

Age. Hettangian–Sinemurian, Early Jurassic (Harris, 1957; Kermack, 1975; Kermack, Mussett & Rigney, 1981).

Locality. Pontalun quarry, near Bridgend in South Glamorgan, Wales, UK (Evans, 1980).

Stratigraphic horizon. Lower Liassic fissure infills (Evans, 1980).

Holotype. UCL T.1503: right dentary.

Referred material. More than 1,000 specimens housed in the UCL (see Evans, 1980; Evans, 1981).

Diagnosis. Early rhynchocephalian distinguished from other diapsids on the basis of the following unique combination of character-states: incomplete lower temporal bar and a fixed quadrate; frontals and parietals unpaired; no supratemporal; reduced lacrimal; postfrontal and postorbital discrete elements; exoccipitals normally fused to basioccipital; no fenestra rotunda; quadrate with broad median lamina; quadratojugal reduced and fused to quadrate conch; quadrate foramen present; quadrate head supported in a ventromedial flange of the squamosal; squamosal large and tetradiate with a ventral process holding quadrate; premaxillae paired; well developed palatal dentition but no teeth on parasphenoid or pterygoid flange; dentition pleurodont; dentary has posterior process articulating with articular complex; dentary closes Meckelian fossa in midregion; low coronoid process; partial fusion of articular, surangular and angular; no splenial; amphicoelous notochordal vertebrae; primitive zygosphene/zygantrum; intercentra present throughout column; single headed ribs on all presacral vertebrae except first few cervicals; atlas and axis with ribs; caudal fracture planes; T-shaped interclavicle; scapula and coracoid fused to form a solid plate; humerus with both ectepi- and entepicondylar foramina; pelvic girdle with large thyroid fenestra; large fourth distal tarsal; hooked fifth metatarsal; gastralia present (Evans, 1980: 204, 205).

Remarks. Gephyrosaurus bridensis is currently considered as the sister-taxon of all remaining rhynchocephalian lepidosauromorphs (Evans & Jones, 2010) and, as a result, it is a key taxon to sample early lepidosauromorph morphological diversity. As for Planocephalosaurus robinsonae, this species has been used as a representative of Lepidosauromorpha in previous higher-level phylogenetic analyses (e.g., Senter, 2004; Pritchard et al., 2015). Gephyrosaurus bridensis was described in detail by Evans (1980) and Evans (1981).

Cteniogenys Gilmore, 1928, Cteniogenys sp. (sensu Evans, 1989)

Age. Late Bathonian, Middle Jurassic (Freeman, 1976; Evans, 1990).

Locality. The Old Cement Works quarry, Kirtlington, Oxfordshire, UK (Evans, 1989).

Stratigraphic horizon. Kirtlington Mammal Bed, near the base of the Forest Marble (Evans, 1989).

Material. More than a hundred of mainly isolated cranial and postcranial bones housed in the NHMUK PV (see Evans, 1990; Evans, 1991).

Diagnosis. The genus Cteniogenys was diagnosed by Evans (1989: 582) by the following features: dentary with double lateral rows of sensory foramina with posteriorly directed grooves; dentary with dorsal and ventral splenial facets that taper anteriorly and end well behind the symphysis, leaving the anterior part of the Meckelian fossa opening medially; medially directed symphysis of the lower jaw with the bulk of the symphysial surface along the upper margin of the Meckelian fossa; subthecodont tooth implantation; broad conical teeth with striations confined to the upper part of the crown and more prominent lingually.

Remarks. Cteniogenys antiquus is the type species of the genus and is known from the Upper Jurassic Morrison Formation of Wyoming, USA (Gilmore, 1928). A second species, Cteniogenys reedi, was described from the Late Jurassic of Guimarota, Portugal (Seiffert, 1970; Seiffert, 1973), but was considered more recently a junior synonym of the type species (Estes, 1983). Subsequently, cranial and postcranial remains from a Middle Jurassic microvertebrate site in Oxfordshire, England, were referred to Cteniogenys sp. (Evans, 1989; Evans, 1990; Evans, 1991).These specimens were left unassigned at the species level because of the age difference of at least 15 million years between the British specimens and those from Portugal and North America (Evans, 1989). Most of the cranial and posctcranial anatomy of the Cteniogenys material from Oxfordshire is known and has been described by Evans (1990) and Evans (1991). The scorings of Cteniogenys sp. in the present phylogenetic analysis are based only on the Middle Jurassic specimens from Oxfordshire.

Simoedosaurus lemoinei Gervais, 1877

Age. Thanetian, late Paleocene (Sigogneau-Russell, 1985).

Locality. Mouras Quarry, Cernay-Berru, Marne, France (Sigogneau-Russell, 1985).

Neotype. MNHN BR 1935: left half of a dorsolaterally crushed skull that lacks the braincase.

Referred material. Multiple specimens housed in the MNHN and SMNS collections (see Sigogneau-Russell & Russell, 1978; Sigogneau-Russell, 1981)

Diagnosis. Medium-sized choristoderan that differs from other diapsids in the following combination of features: snout narrow and represents about 40% of the total length of the skull; internal choanae situated in the posterior half of the snout; lacrimal short; contact between postorbitofrontal and parietal situated approximately at the level of the anterior edge of the supratemporal fenestra; premaxillary teeth very large; maxillary teeth large anteriorly and very small at the posterior end of the tooth row (Sigogneau-Russell, 1985: 766).

Remarks. Simoedosaurus lemoinei is the only post-Jurassic terminal included in the taxonomic sample of this phylogenetic analysis. It was chosen to represent the morphology of choristoderans because it is one of the best known and described members of this enigmatic clade of neodiapsids. Simoedosaurus lemoinei was described in detail by Sigogneau-Russell & Russell (1978) and Sigogneau-Russell (1981).

Aenigmastropheus parringtoni Ezcurra, Scheyer & Butler, 2014

Age. Middle–late Wuchiapingian, middle late Permian (Angielczyk et al., 2014).

Locality. Locality B35 of Stockley, close to the road near Ruanda, Songea District, Ruhuhu Valley, southern Tanzania (Ezcurra, Scheyer & Butler, 2014).

Stratigraphic horizon. Usili Formation (formerly Kawinga Formation), Songea Group, Ruhuhu Basin (Ezcurra, Scheyer & Butler, 2014).

Holotype. UMZC T836: partial postcranial skeleton including five posterior cervical–anterior dorsal vertebrae, distal half of the right humerus, fragment of probable left humeral shaft, proximal end of the right ulna, and three indeterminate fragments of bone (one of which may represent part of a radius) (Ezcurra, Scheyer & Butler, 2014).

Diagnosis. Archosauromorph distinguished from other diapsids on the basis of the following unique combination of character-states (autapomorphy marked with an asterisk): posterior cervical and anterior dorsal vertebrae notochordal, with well-developed anterior and posterior centrodiapophyseal and prezygodiapophyseal laminae, and sub-triangular neural spines in lateral view; humerus with a strong diagonal ridge on the anterior surface of the shaft*; humerus with strongly developed capitellum (radial condyle) and trochlea (ulnar condyle) and without entepicondylar and ectepicondylar foramina; ulna with strongly developed olecranon process forming a single ossification with the rest of the bone (Ezcurra, Scheyer & Butler, 2014: 15, 16).

Remarks. Parrington (1956) described the remains of what he considered “an enigmatic reptile” from the Lopingian of Tanzania. He concluded that this specimen (UMZC T836) did not bear close resemblances to any known synapsid, and suggested instead that the specimen might have close affinities with archosaurs because of the vertebral morphology and the presence of hollow limb bones and an ectepicondylar groove on the humerus. Subsequently, Hughes (1963) noted that the vertebrae of UMZC T836 were not as archosaurian in appearance as Parrington originally thought and that laminae on the neural arch also occur in pelycosaurian synapsids. Gower & Sennikov (2000) noted that UMZC T836 is probably indeterminate, but could possibly be archosaurian. Most recently, Ezcurra, Butler & Gower (2013) indicated that UMZC T836 is likely not referable to Archosauriformes. Ezcurra, Scheyer & Butler (2014) redescribed in detail the anatomy of UMZC T386 and considered that it possessed a unique combination of apomorphies that allowed the erection of a new genus and species: Aenigmastropheus parringtoni. These authors included Aenigmastropheus parringtoni for the first time in a quantitative phylogenetic analysis and recovered it at the base of Archosauromorpha, within a clade composed of Protorosaurus speneri and tanystropheids. Nevertheless, Ezcurra, Scheyer & Butler (2014) stated that further tests on the phylogenetic position of this species should be conducted in the future with an improved character and taxonomic sample of early archosauromorphs. Indeed, Ezcurra, Scheyer & Butler (2014) pointed out that the presence of notochordal vertebrae may place Aenigmastropheus parringtoni as the most basal known archosauromorph.

Protorosaurus speneri Meyer, 1832

Age. Middle Wuchiapingian, middle late Permian (Legler, Gebhardt & Schneider, 2005; Legler & Schneider, 2008; Ezcurra, Scheyer & Butler, 2014).

Localities. Glücksbrunn, Thuringia, Germany (type locality; Gottmann-Quesada & Sander, 2009). Referred specimens were recovered in several localities within Thuringia and Hesse, central Germany. Protorosaurus also occurs at Middridge and Quarrington quarries near Durham, England, UK (Evans & King, 1993; Gottmann-Quesada & Sander, 2009). For a full list of the German localities, see Gottmann-Quesada & Sander (2009: Table 1).

Stratigraphic horizons. Kupferschiefer Formation, Zechstein Group, basal cycle of the Zechstein (Z1) in Germany (type horizon) and Marl Slate Formation in the UK (Ezcurra, Scheyer & Butler, 2014).

Lectotype. NHMW 1943I4: almost complete skeleton missing the skull (Gottmann-Quesada & Sander, 2009).

Referred material. Multiple fairly complete to fragmentary skeletons housed in several European institutions. The complete hypodigm of Protorosaurus speneri was listed by Gottmann-Quesada & Sander (2009: 137, Table 1).

Emended diagnosis. Archosauromorph distinguished from other diapsids on the basis of the following unique combination of character-states (autapomorphies marked with an asterisk): premaxilla with three tooth positions; frontal-nasal suture transverse; squamosal posterior process extends posterior to the head of the quadrate; surangular-angular suture anteroposteriorly convex ventrally in lateral view; distinct mammillary processes on the lateral surface of the neural spine extend up to the tenth presacral*; middle and distal caudal vertebrae with bifurcated neural spine*; coracoid with large biceps process; ulna with olecranon process as a separate ossification; and femoral shaft diameter distally narrowed.

Remarks. Protorosaurus speneri is by far the best-known Permian archosauromorph, being represented by multiple individuals from marine beds of Europe. A fairly complete skeleton of this species provides the best available evidence of a Permian archosauromorph body plan (Gottmann-Quesada & Sander, 2009; Ezcurra, Scheyer & Butler, 2014). The anatomy of Protorosaurus speneri has been recently described in detail by Gottmann-Quesada & Sander (2009). This species has been alternatively recovered in quantitative phylogenetic analyses within a monophyletic Protorosauria (together with tanystropheids; e.g., Benton, 1985; Dilkes, 1998; Ezcurra, Scheyer & Butler, 2014), as one of the sister-taxa of tanystropheids and more crownward archosauromorphs (Gottmann-Quesada & Sander, 2009), or the most basal archosauromorph (Pritchard et al., 2015). Therefore, despite the rather complete knowledge of the anatomy of Protorosaurus speneri, its phylogenetic position among archosauromorphs is still much debated.

Macrocnemus bassanii Nopcsa, 1930

Age. Late Anisian–early Ladinian, Middle Triassic.

Localities. Besano, Italy (type locality); Monte San Giorgo, Switzerland.

Stratigraphic horizons. Scisti bituminosi, Besano Formation, Italy (type horizon); Alla Cascina Member, Meride Limestone Formation, Switzerland.

Holotype. MSNM specimen Besano I (probably destroyed during WWII): partial skeleton including skull, complete vertebral column, humerus, tibia and fibula.

Referred material. Several specimens housed in the collections of MSNM and PIMUZ (see Peyer, 1937; Rieppel, 1989a).

Emended diagnosis. Small tanystropheid that differs from other archosauromorphs in the following combination of features: anteriorly curved, “U”-shaped suture between frontal and parietal in dorsal view; posterolateral processes of the parietal strongly posterolaterally oriented; marginal teeth recurved, with posteriorly concave distal margin; 24 presacral vertebrae; humerus less than 10% longer than radius; and tibia and fibula at least 20% longer than femur. This diagnosis is composed of the synapomorphies of the genus Macrocnemus found by Pritchard et al. (2015) and the differences reported by Li, Zhao & Wang (2007) and Fraser & Furrer (2013) between Macrocnemus bassanii and Macrocnemus fuyuanensis and Macrocnemus obristi, respectively.

Remarks. Macrocnemus bassanii is a tanystropheid known from several rather complete skeletons that lack the extremely long necks present in more deeply nested members of the group (e.g., Tanystropheus longobardicus, Tanytrachelos ahynis) (Peyer, 1937). Peyer (1937) provided a detail description of the species, which was partially complemented by Rieppel (1989a). Macrocnemus bassanii was repeatedly used as a representative member of Tanystropheidae in phylogenetic analyses (e.g., Dilkes, 1998; Senter, 2004; Gottmann-Quesada & Sander, 2009) and recently has been recovered as the basalmost tanystropheid together with the congeneric species Macrocnemus fuyuanensis (Pritchard et al., 2015).

Amotosaurus rotfeldensis Fraser & Rieppel, 2006

Age. Early Anisian, early Middle Triassic.

Locality. Kossig quarry, Rotfelden, in district of Calw, Germany (Fraser & Rieppel, 2006).

Stratigraphic horizon. Röt Formation, ‘Violet horizon 5’ of Ortlam (1967), Upper Buntsandstein (Fraser & Rieppel, 2006).

Holotype. SMNS 50830: partial skull, cervical series, scattered dorsal vertebrae, and pectoral and pelvic elements (Fraser & Rieppel, 2006).

Referred material. Multiple slabs preserving partially articulated cranial and postcranial bones, including SMNS 50691, 54783, 54784a, 54784b, 54810, 90540, 90543, 90544, 90552, 90559, 90563, 90564, 90566, 90599–90601 and several unnumbered specimens.

Diagnosis. Amotosaurus rotfeldensis is a small tanystropheid that was diagnosed by Fraser & Rieppel (2006: 867) on the basis of the following features: eight cervical vertebrae; the centra of cervicals 4 and 5 are the longest and are at least 2.5 times as long as their minimum height; elongate cervical ribs extending across at least three intervertebral articulations anteriorly; 25 presacral vertebrae; second sacral rib distinctly bifurcate; length of metatarsals asymmetric with IV being the longest, then III, then II, then I and V being the shortest; proximal phalanx on digit V elongate and ‘metatarsal-like’; three distal tarsals in the ankle; the ischium and pubis almost touch below the level of the thyroid fenestra; the vomers, palatines and pterygoids are all covered by a fine shagreen of denticles.

Remarks. Wild (1980) interpreted multiple tanystropheid specimens from the Middle Triassic of the Black Forest (southwest Germany) as juveniles of “Tanystropheus” antiquus (Wild, 1980), but this hypothesis was subsequently questioned by several authors who suggested that they may belong to a new genus (e.g., Evans, 1988; Wild, 1987). Fraser & Rieppel (2006) revised the taxonomy of these specimens and concluded that they belong to a new genus and species, Amotosaurus rotfeldensis. Fraser & Rieppel (2006) thus considered the new species distinct from “Tanystropheus” antiquus, which was subsequently transferred to the new genus Protanystropheus by Sennikov (2011), resulting in the new combination Protanystropheus antiquus. Fraser & Rieppel (2006) described briefly the anatomy of Amotosaurus rotfeldensis and a detailed description of the species is needed. Based on limited character data, Pritchard et al. (2015) found Amotosaurus rotfeldensis as more closely related to Tanystropheus longobardicus than to other sampled tanystropheids. My examination of available specimens of Amotosaurus rotfeldensis allowed me to score a vast number of characters that were not described or figured in the original description of the species. However, a detailed description of Amotosaurus rotfeldensis would require further preparation of multiple specimens and is beyond the scope of the present paper.

Tanystropheus longobardicus (Bassani, 1886)

Age. Late Anisian–early Ladinian, Middle Triassic.

Localities. Besano, Varese Province, Lombardy, Italy (type locality; Bassani, 1886); Monte San Giorgo, Valporina, Switzerland (neotype locality; Wild, 1973).

Stratigraphic horizons. Besano Formation, Italy (type horizon; Bassani, 1886); Meride Formation, Switzerland (neotype horizon; Wild, 1973).

Neotype. PIMUZ T 2791: fairly complete and partially articulated skeleton of a probably young individual that lacks the distal half of the tail (Wild, 1973).

Referred material. Dozens of juvenile and adult specimens housed in the collections of MSNM, PIMUZ and SMNS (see lists of specimens in Wild, 1973; Nosotti, 2007).

Emended diagnosis. Large tanystropheid that differs from other archosauromorphs in the following combination of features: frontals flared laterally as wing-like structures above the orbits; large pineal foramen enclosed beween frontals and parietals; ventrally flexed anterior end of dentary; strongly posteriorly developed retroarticular process of the lower jaw; conical and straight marginal tooh crowns with longitudinal ridges; 13 cervical vertebrae; length of the centra of the fourth and fifth cervical vertebrae at least 14 times their heights; distal end of second sacral rib not bifurcated; two ossified distal carpals; and manual digit IV composed of four phalanges.

Remarks. The genus Tanystropheus and species Tanystropheus conspicuus were erected by Meyer (1847–1855) based on isolated bones from the European Upper Muschelkalk, which were interepreted as strongly elongated caudal vertebrae of a reptile. Subsequently, Bassani (1886) named the new genus and species Tribelesodon longobardicus based on cranial and postcranial remains from the Middle Triassic of Besano (Italy), interpreting it as a flying reptile (Nopsca, 1923). Peyer (1931) described new specimens of Tanystropheus from Monte San Giorgo (Switzerland) and reinterpreted the elongated type bones of Tanystropheus conspicuus as cervical vertebrae. In addition, Peyer (1931) proposed that Tribelesodon and Tanystropheus were cogeneric, referred all the specimens from Besano and Monte San Giorgio to Tanystropheus longobardicus, and designated Tanystropheus longobardicus as the type species of the genus. Peyer (1931) reinterpreted Tanystropheus as a terrestrial reptile with an extremely long neck adapted to catch prey from the shore of water bodies. Wild (1973) provided a detailed description of Tanystropheus longobardicus, and the anatomical knowledge of the species was further improved by Nosotti (2007) with the description of new specimens. Tanystropheus longobardicus has been commonly included in the taxonomic sampling of phylogenetic analyses focused on early archosauromorph interrelationships and represents the best-known tanystropheid (e.g., Benton, 1985; Dilkes, 1998; Gottmann-Quesada & Sander, 2009; Ezcurra, Scheyer & Butler, 2014; Pritchard et al., 2015).

Jesairosaurus lehmani Jalil, 1997

Age. Late Olenekian–Anisian, Early to Middle Triassic (Lehman, 1971; Jalil & Taquet, 1994).

Locality. Site 5003 of Busson, east Algeria (Jalil, 1997).

Stratigraphic horizon. Lower sandstones of the lower Zarzaïtine Formation, Zarzaitine Series (Jalil, 1997).

Holotype. ZAR 06: skull, neural arches of the last five cervical vertebrae, pectoral girdle and proximal end of the left humerus (modified from Jalil, 1997).

Referred material. ZAR 07: partial, dorsoventrally compressed skull; ZAR 08: partial skull and postcranium; ZAR 09: two partial postcranial skeletons preserved in the same block (ZAR 09A is a skull originally articulated with one of the postcraniums of ZAR 09 and prepared by serial grinding); ZAR 10: poorly preserved scapula and at least four dorsal vertebrae; ZAR 11: pelvic girdle, at least three dorsal and three caudal vertebrae and some gastralia; ZAR 12: poorly preserved pelvic girdle and at least five dorsal and three caudal vertebrae; ZAR 13: seven articulated presacral vertebrae (ZAR 13A is a pelvic girdle originally articulated to ZAR 13 and prepared by serial grinding); ZAR 14: pelvic girdle; and ZAR 15: pelvic girdle and partial hindlimb (modified from Jalil, 1997).

Emended diagnosis. Small archosauromorph that differs from other diapsids in the following combination of features: posterior process of the maxilla well developed and forming most of the ventral border of the orbit; extensive contact between palatine and maxilla; ascending process of the jugal well developed extending posteriorly to the posterior border of the orbit and contacting or lying close to the squamosal; postfrontal subequal in size to the postorbital; small pineal foramen restricted to and enclosed by the anterior end of the parietals; palatal teeth not arranged in distinct rows; neck anteroposteriorly shorter than the skull; basioccipital with a faint median keel on the ventral surface; middle and posterior dorsal vertebrae with anteroposteriorly or posteriorly expanded distal ends of the neural spines; scapulacoracoid inverted L-shape in lateral view, with a strongly posteriorly developed coracoid; humerus with apparently fully closed ectepicondylar foramen; pubis and ischium of the hemipelvis fused to each other; and hindlimb longer than forelimb and relatively large in comparison with the rest of the postcranial skeleton (modified from Jalil, 1997: 508).

Remarks. Jesairosaurus lehmani was described in detail by Jalil (1997). Despite its short neck, this species has been considered since its original description as a member of “Prolacertiformes.” Nevertheless, the phylogenetic position of this species has not been further tested in more recent quantitative analyses. Some comments on the anatomy of the species are added here that are informative for the phylogenetic analyses conducted here. The holotype specimen (ZAR 06) is a partial skull with articulated lower jaw. In this specimen, the anterior end of the dentary and its symphysis are complete and, as a result, the premaxillae should be fairly complete. The right premaxilla preserves six teeth in place, but there is room in the alveolar margin of the bone for nine to ten tooth positions (ZAR 06), resembling the high premaxillary tooth count present in some basal saurians (e.g., Gephyrosaurus bridensis: Evans, 1980). The anterodorsal margin of the maxilla is strongly concave and there is no evidence for a facet for reception of a postnarial process of the premaxilla. The lateral surface of the anterior process of the maxilla possesses a large, oval foramen at the level of the third maxillary tooth, which seems to be homologous with the anterior maxillary foramen of other saurians (e.g., Planocephalosaurus robinsonae: NHMUK PV R9954; Protorosaurus speneri: Gottmann-Quesada & Sander, 2009; Prolacerta broomi: Modesto & Sues, 2004). A row of smaller neurovascular foramina extends posteriorly, posterior to the level of the third maxillary tooth position. A total of 20 or 21 tooth positions are estimated in the maxillae of ZAR 06. The maxillary tooth crowns are straight, with convex mesial and distal margins in labial view. The teeth are not fused to the tooth bearing bones and they possess long roots, indicating that they were deeply implanted in the sockets (ZAR 06, 09). In one specimen (ZAR 09) there is a distinct medial wall to the alveoli and, as a result, the tooth implantation was probably subthecodont.

Figure 6 Jesairosaurus lehmani.

Partial skull of a referred specimen (ZAR 07) in dorsal view, and close up of the pineal foramen and frontal–parietal suture. Numbers indicate character-states scored in the data matrix and the arrow indicates anterior direction. Abbreviations: ax, axis; fr, frontal; or, orbit; pfr, prefrontal; po, postorbital; sq, squamosal; stf, supratemporal fenestra. Scale bar equals 5 mm.

The quadrate is shallowly emarginated posteriorly and lacks the distinct lateral projection of its anterior magin that characterizes the quadrate conch of lepidosauromorphs (Gauthier, Kluge & Rowe, 1988). The pineal foramen is small and oval, with an anteroposterior main axis, and is mostly or probably completely enclosed by both parietals (contra Jalil, 1997) (Fig. 6). The parietal lacks a dorsal emargination on the posterior margin of the posterolateral process (ZAR 06, 08), contrasting with the condition in other basal archosauromorphs (Müller, 2004). The paroccipital process of the opisthotic is laterally well developed and contacts extensively with the posterolateral process of the parietal. As a result, the posttemporal fenestra was very small, if it was present. The anterior end of the dentary curves gently ventrally and medially, resembling the condition in Gephyrosaurus bridensis (Evans, 1980) and Protorosaurus speneri (Gottmann-Quesada & Sander, 2009). The lateral surface of the dentary of ZAR 06 possesses four neurovascular foramina aligned in a mainly longitudinal row. There is at least one posterior dentary tooth that possesses a gentle mesiodistal constriction between the crown and the root in ZAR 08.

A total number of nine cervical vertebrae can be estimated based on the position of the pectoral girdle with respect to the axial skeleton in ZAR 06. All the neural spines exposed in ZAR 06 are anterodorsally oriented, as occurs in the eighth and ninth cervical vertebrae of ZAR 08. By contrast, the probable sixth and seventh cervicals of ZAR 08 possess vertical neural spines. This intraspecific variation in the orientation of the cervical neural spines resembles that present among specimens of Proterosuchus fergusi (GHG 231, SAM-PK-11208, K140). There is no fossa immediately lateral to the base of the neural spine in the cervicals of ZAR 08 and the cervical neural spines lack a transverse distal expansion or mammillary processes in ZAR 06 and ZAR 08. The seventh cervical neural spine of ZAR 08 possesses an anteroposteriorly expanded distal end, with an acute anterior projection.

The dorsal series is represented by 15 vertebrae in ZAR 08. There is no evidence of laminae on the dorsal neural arches in the available specimens (e.g., ZAR 11, 12). The dorsal vertebrae lack fossae immediately lateral to the base of the neural spine in two specimens (ZAR 08, 09), but they are present as deep, subcircular pits in another specimen (ZAR 13). The neural spines of the middle dorsal vertebrae possess a strong posterior projection of their distal end (ZAR 08), whereas an anterior projection is variable in the two individuals assessed as ZAR 09. The distal ends of the neural spines may have contacted each other and lack a transverse expansion of the distal and mammillary processes (ZAR 09, 13). Broken dorsal centra show an internal structure composed of trabeculae, but there is no large, central opening that would indicate the presence of a notochordal canal (ZAR 13). The same condition was observed in the postcranial axial series of other specimens and, as a result, the vertebrae of Jesairosaurus lehmani are reinterpreted here as not notochordal (contra Jalil, 1997).

Three probable sternal plates are preserved immediately posterior to the coracoids and posterolateral to the posterior ramus of the interclavicle in ZAR 09. The right sternal plate is preserved as a mould and the two left elements are preserved as poorly mineralized elements aligned anteroposteriorly to each other. The sternal plates are oval, with a transversely oriented main axis. Two of the sternal plates are paired in the transverse plane and seem to have had a median longitudinal contact. As described by Jalil (1997), the ectepicondylar foramen of the humerus appears to have been fully closed (ZAR 09).

The proximal articular surface of the femur is convex in ventral view, suggesting a rather well ossified head (ZAR 14), contrasting with the flat or concave and poorly ossified proximal end present in rhynchosaurs (e.g., Mesosuchus browni: SAM-PK-7416; Stenaulorhynchus stockleyi: Huene, 1938), Prolacerta broomi (BP/1/2675), Proterosuchus fergusi (SAM-PK-140) and erythrosuchids (e.g., Erythrosuchus africanus: NHMUK PV R3592). The presence of an internal or fourth trochanter is equivocal (ZAR 14) and the distal end of the femur does not taper distally in side view (ZAR 15), contrasting with the condition in Protorosaurus speneri (SMNS 55387, cast of Simon/Bartholomäus specimen) and tanystropheids (Amotosaurus rotfeldensis: SMNS 54783). The proximal tarsals were not described by Jalil (1997), but they are present in ZAR 15, although poorly preserved, and it can be at least determined that the calcaneum lacks a calcaneal tuber. It is not possible to assess the presence or absence of a perforating foramen between the proximal tarsals.

Pamelaria dolichotrachela Sen, 2003

Age. Anisian, early Middle Triassic (Lucas, 2010).

Localities. Three adjacent sites about 4 kilometres north–northwest of the Yerrapalli village, Adilabad district, Andhra Pradesh, India (Sen, 2003).

Stratigraphic horizon. Yerrapalli Formation, Gondwana Supergroup, Pranhita-Godavari Basin (Sen, 2003).

Holotype. ISI R316: partial cranial and postcranial skeleton (Sen, 2003).

Referred material. ISI R317: partial cranial, presacral series and some appendicular bones; ISI R318–333: isolated bones found in association with the holotype of Yarasuchus deccanensis (Sen, 2003; Sen, 2005).

Diagnosis. Pamelaria dolichotrachela is a medium-sized basal archosauromorph that was diagnosed by Sen (2003: 664) on the basis of the following features: external naris small and confluent; vomer posteriorly wide; ventrally directed plate-like process of prootic anterior to ventral ramus of opisthotic; coronoid process prominent; dentary with approximately 19 teeth; and additional spinous projection placed anteriorly with abrupt shift in the position of neural spine in the posterior caudals.

Remarks. Sen (2003) originally erected and described Pamelaria dolichotrachela as a prolacertiform archosauromorph. More recently, Nesbitt et al. (2015) recovered this species as the most basal member of Allokotosauria. Sen (2003) provided a good account of the anatomy of the species, but a revised description and updated comparisons with other allokotosaurians would improve the anatomical knowledge of Pamelaria dolichotrachela. A detailed revision of the anatomy of this species is beyond the scope of this contribution and is currently in preparation by the author and colleagues.

Azendohsaurus madagaskarensis Flynn et al., 2010

Age. Late Ladinian–early Carnian, late Middle Triassic–early Late Triassic (Nesbitt et al., 2015).

Locality. Drainage of the Malio River, southwestern Madagascar (Flynn et al., 2010; Nesbitt et al., 2015).

Stratigraphic horizon. Basal ‘Isalo II’ of Besairie (1972), termed the Makay Formation by Razafimbelo (1987); Morondava Basin (Flynn et al., 2010; Nesbitt et al., 2015).

Holotype. UA 7-20-99-653 (field number 7-20-99-653): a nearly complete skull with associated vertebrae (Flynn et al., 2010).

Paratypes. FMNH PR 2751 (field number 8-30-98-376), nearly complete disarticulated skull (associated with postcranial specimens FMNH PR 2788, FMNH PR 2789, FMNH PR 2792 and possibly FMNH PR 2796) (Nesbitt et al., 2015).

Referred material. Around 300 specimens that preserve cranial and postcranial bones and were listed by Nesbitt et al. (2015: Appendix 1).

Diagnosis. Azendohsaurus madagaskarensis is a medium-sized (2–3 m in length), early-diverging archosauromorph that differs from all other archosauromorphs in possessing the following unique combination of character-states: ventral curvature of the anterior portion of the dentary; a robust dorsal process of the maxilla, the base of which occurs on the anterior third of the bone; a concave anterior margin of the dorsal process of the maxilla; lanceolate teeth with denticles; a series of small nutrient foramina on the medial surface of the maxilla; elongated cervical vertebrae with small epipophyses dorsal to the postzygapophyses; small tuber located on the ventrolateral surface of the prezygapophyseal stalk in the middle to posterior cervical vertebrae; deep fossae between well developed laminae in the posterior cervical vertebrae; hyposphene-hypantra intervertebral articulations in the posterior cervical, anterior trunk, and sacral vertebrae; well-defined fossa at the base of the neural spine, just posterior to the prezygapophyses in the second sacral vertebra; oval and proximodistally oriented tuber on the lateral surface of the scapula that nearly contacts the edge of the glenoid fossa; posteriorly expanded, T-shaped interclavicle; lateral side of the calcaneal tuber expanded laterally and ventrally, with the ventral expansion being clearly visible in proximal view; and proximal projection on the proximal surface of metatarsal IV (Flynn et al., 2010; Nesbitt et al., 2015).

Remarks. Azendohsaurus laaroussii was named by Dutuit (1972) based on teeth and tooth-bearing elements from the Argana Formation of Morocco. Dutuit (1972) identified Azendohsaurus laaroussii as an ornithischian dinosaur, but it subsequently was reidentified as a sauropodomorph dinosaur (Thulborn, 1973; Bonaparte, 1976; Gauffre, 1993; Flynn et al., 1999). Flynn et al. (2010) named the new species Azendohsaurus madagaskarensis from the Makay Formation of Madagascar and the remains of this species provided for the first time a comprehensive knowledge of the cranial anatomy of the genus. This new information allowed Flynn et al. (2010) to reinterpret Azendohsaurus laaroussii as a non-archosaurian archosauromorph rather than an herbivorous dinosaur. More recently, Nesbitt et al. (2015) described in detail the postcranial anatomy of Azendohsaurus madagaskarensis and, as a result, this species is currently one of the best-known early archosauromorphs. Nesbitt et al. (2015) included both species of Azendohsaurus in a phylogenetic analysis and recovered them as members of the new clade Allokotosauria, which was found as the sister-taxon of the clade composed of Prolacerta broomi and Archosauriformes.

Trilophosaurus buettneri Case, 1928

Age. Middle Norian to possibly Rhaetian; late Late Triassic (Parker & Martz, 2011).

Localities. Near Walker’s Tank, Texas, USA (type locality; Case, 1928); multiple localities that belong to several late Upper Triassic formations in Texas, New Mexico and Arizona, southwest USA (Long & Murry, 1995; Spielmann et al., 2008).

Stratigraphic horizons. Tecovas Formation (type horizon; Case, 1928) and several other late Upper Triassic formations (see Long & Murry, 1995; Spielmann et al., 2008).

Holotype. UMMP 2338: an incomplete right dentary fragment bearing parts of five teeth.

Referred material. Hundreds of specimens composed of cranial and postcranial remains that were listed by Spielmann et al. (2008: Appendix 1).

Diagnosis. Spielmann et al. (2008: 11) diagnosed the genus Trilophosaurus as an archosauromorph that can be distinguished from all other archosauromorphs by its transversely broad, tricuspid teeth and a femur with a prominent internal trochanter that extends one-third of the way down the shaft. In addition, Spielmann et al. (2008: 11) distinguished Trilophosaurus buettneri from Trilophosaurus jacobsi by the lack of prominent cingula linking the cusps both labiolingually across the center of the tooth and also along the mesial and distal margins of the tooth; central cusp subequal in height to the labial and lingual cusps; central cusp not displaced labially or lingually, so tooth crown is labiolingually symmetrical in occlusal view; skull possessing a single parasaggital crest; cervical vertebrae with bifurcate postzygapophyses; procoelous cervical centra; double keeled sacral centra; a lack of prominent ridges on the sacral vertebrae extending from the posterior margin of the pre- and postzygapophyses to the base of the neural spine; sacral neural spines extend nearly the entire length of the centra; rectangular ectepicondyle; radial condyle of humerus larger than ulnar condyle; proximal femur is rhombus-shaped; astragalus with pointed calcaneal articular surface; ridge developed on posterior astragalus; and “neck” of astragalus gracile and elongate.

Remarks. Trilophosaurus buettneri was named by Case (1928) based on a partial dentary bearing labiolingually broad, tricuspid teeth. Subsequently, more complete specimens and partial skeletons were collected from the Triassic of the southwest USA and Gregory (1945) described in detail the anatomy of the species. The cranial anatomy of Trilophosaurus buettneri was revised by Parks (1969) and Spielmann et al. (2008) redescribed and figured comprehensively the anatomy of the species. More recently, Nesbitt et al. (2015) reinterpreted and redescribed the carpal and hand anatomy of Trilophosaurus buettneri. This species has been frequently included in phylogenetic analyses focused on early archosauromorphs, but its position among the main lineages of the group has been a matter of debate. Trilophosaurus buettneri has been alternatively recovered as one of the most basal archosauromorphs (Benton, 1985), the sister-taxon of rhynchosaurs (Gottmann-Quesada & Sander, 2009), the sister-taxon of the clade composed of rhynchosaurs, Prolacerta broomi and archosauriforms (Dilkes, 1998; Ezcurra, Scheyer & Butler, 2014), or the sister-taxon of Prolacerta broomi and archosauriforms (Pritchard et al., 2015). More recently, Nesbitt et al. (2015) recovered Trilophosaurus buettneri as closely related to Azendohsaurus within the new clade Allokotosauria. Trilophosaurus buettneri and other allokotosaurians were recovered as the sister-taxon of Prolacerta broomi and archosauriforms by Nesbitt et al. (2015).

Noteosuchus colletti Watson, 1912a

Age. Induan, earliest Triassic (Carroll, 1976).

Locality. Near Grassy Ridge, Eastern Cape Province, South Africa (Watson, 1912a; Carroll, 1976).

Stratigraphic horizon. Base of the Lystrosaurus AZ, Katberg Formation, Tarkastad Subgroup, Beaufort Group, Karoo Supergroup, Karoo Basin (Watson, 1912a; Carroll, 1976).

Holotype. AM 3591: partial postcranium, missing the neck, pectoral girdle, right forelimb and distal half of the tail. The vast majority of the bones of the specimen were mechanically destroyed in order to study their natural moulds (Carroll, 1976).

Emended diagnosis. Small rhynchosaur that differs from other basal archosauromorphs in the following unique combination of features: second sacral rib bifurcated distally, with a tapering posterolateral process (squared in Mesosuchus browni); anterior caudal vertebrae with a neural spine 2.1 times dorsoventrally taller than their anteroposterior length at base (ratio equals 3.0–3.5 in Mesosuchus browni); manual unguals considerably longer than preungual phalanges; ilium without preacetabular process (small preacetabular process in Mesosuchus browni) and postacetabular process mainly posteriorly oriented (posterodorsally oriented in Mesosuchus browni); and length of metatarsal I around 0.40 times the length of metatarsal III.

Remarks. Dilkes (1998) suggested that Noteosuchus colletti is potentially a subjective junior synonym of the rhynchosaur Mesosuchus browni from the early Anisian of South Africa. This potential synonymy was based on the presence of a consistent morphology between both species and the shared features of a median ventral groove on the centra of the first two caudal vertebrae and a flattened first distal tarsal (Dilkes, 1998). Noteosuchus colletti has been largely ignored in discussions of rhynchosaur evolution (e.g., Benton, 1983; Benton, 1990; Wilkinson & Benton, 1995; Hone & Benton, 2008; Montefeltro et al., 2013; Mukherjee & Ray, 2014). Ezcurra, Scheyer & Butler (2014) scored Noteosuchus colletti and Mesosuchus browni as independent terminals in their phylogenetic analysis because the temporal gap between the two species apparently spans most of the Early Triassic (ca. 5 million years) and the former species could potentially shed light on the minimal divergence time of Rhynchosauria. Ezcurra, Scheyer & Butler (2014) recovered Noteosuchus colletti and Mesosuchus browni within Rhynchosauria in a trichotomy with Howesia browni, and a consistent result was recovered by Ezcurra, Montefeltro & Butler (2016) using an independent dataset. Ezcurra, Montefeltro & Butler (2016) revisited some anatomical features of Noteosuchus colletti and highlighted differences between the sacral and caudal vertebrae and ilium of this species and those of Mesosuchus browni (e.g., tapering posterolateral process of the distally bifurcated second sacral rib, proportionally lower neural spines of the anterior caudal vertebrae, absence of preacetabular process and posterodorsally oriented postacetabular process on the ilium), which support the hypothesis that Noteosuchus colletti is a valid taxon.

Mesosuchus browni Watson, 1912b

Age. Early Anisian, early Middle Triassic (Rubidge, 2005).

Locality. Site along a road between Aliwal North and Lady Grey (exact location of the site unknown), Eastern Cape Province, South Africa (Dilkes, 1998).

Stratigraphic horizon. Burgersdorp Formation, Cynognathus AZ subzone B, Tarkastad Subgroup, Beaufort Group, Karoo Supergroup, Karoo Basin (Dilkes, 1998).

Holotype. SAM-PK-5882: partial snout, palate, braincase, lower jaws, sections of articulated presacral vertebral column, nine articulated caudal vertebrae, portions of scapula and pelvic girdle, and partial forelimb and hindlimbs (Dilkes, 1998).

Referred material. SAM-PK-6046: incomplete right maxilla, an articulated series of the last ten presacrals, both sacrals, and first six caudals, partial forelimbs, left and right pelvic girdles, right hindlimb, elements of left tarsus; SAM-PK-6536: virtually complete skull with lower jaws, articulated cervical vertebrae and ribs, dorsal vertebrae and ribs, complete left scapulocoracoid and partial right scapula, interclavicle, clavicles, distal end of left humerus, and gastralia; SAM-PK-7416: an articulated vertebral column composed of the last dozen presacrals, both sacrals and at least the first 15 caudal vertebrae, fragments of right forelimb, pelvic girdle, complete right femur, right crus and partial left crus, and right and left tarsi and pedes (Dilkes, 1998).

Diagnosis. Dilkes (1998: 503) diagnosed Mesosuchus browni on the basis of the following autapomorphies: multiple rows of maxillary and dentary teeth with each row consisting of only a very small number of teeth; two premaxillary teeth that are approximately twice the size of the maxillary teeth; maxillary teeth inset medially and project below the internal naris; occlusion between vomerine teeth and dentary teeth; saddle-shaped vomers that overhang dorsally the premaxillary symphysis; length of axis neural spine greater than length of axis centrum; anteroposteriorly narrow neural spine of third cervical; and prominent midventral groove on first two caudal centra.

Remarks. Mesosuchus browni is the best-known non-rhynchosaurid rhynchosaur species (Dilkes, 1998) and has been used as a representative of rhynchosaur morphology in higher-level phylogenetic analyses focused on archosauromorphs (e.g., Dilkes, 1998; Gottmann-Quesada & Sander, 2009; Ezcurra, Scheyer & Butler, 2014; Pritchard et al., 2015) and to root the comprehensive phylogenetic analysis focused on early archosauriforms of Nesbitt (2011). Mesosuchus browni possesses an intermediate morphology between that of non-rhynchosaurian basal archosauromorphs and that of later rhynchosaurs, lacking, for example, the distinct blade and groove dental apparatus that is characteristic of rhynchosaurids (Dilkes, 1998). This species was described in detail by Dilkes (1998).

Howesia browni Broom, 1905

Age. Early Anisian, early Middle Triassic (Rubidge, 2005).

Locality. Unknown locality near the town of Aliwal North, Eastern Cape Province, South Africa (Dilkes, 1995).

Stratigraphic horizon. Burgersdorp Formation, Cynognathus AZ subzone B, Tarkastad Subgroup, Beaufort Group, Karoo Supergroup, Karoo Basin (Dilkes, 1995).

Holotype. SAM-PK-5884: partial skull with palate and incomplete lower jaws (Dilkes, 1995).

Referred material. SAM-PK-5885: dorsoventrally crushed skull with partial palate, braincase and atlas-axis complex; SAM-PK-5886: partial, articulated postcranium consisting of the posterior four dorsal, both sacral and first dozen caudal vertebrae, incomplete pelvic girdle, partial left hindlimb, and complete right tarsus (Dilkes, 1995).

Diagnosis. Dilkes (1995: 666) diagnosed Howesia browni as a small rhynchosaur diapsid characterized by the following autapomorphies: multiple rows of small, conical teeth with ankylothecodont implantation in medially expanded maxillae that lack longitudinal, occlusal grooves; multiple rows of numerous conical teeth on the dentaries; broad ventral process of squamosal that does not extend below middle of infratemporal fenestra; horizontal shelf on medial side of quadrate ramus of pterygoid; contact between ectopterygoid and jugal reduced to less than half of the length of the distal expansion of the ectopterygoid; deep pockets on neural arches of posterior dorsal and sacral vertebrae; and tall, posteriorly inclined neural spines of proximal caudal vertebrae.

Remarks. Howesia browni is a non-rhynchosaurid rhynchosaur from the Cynognathus AZ that is less well known than Mesosuchus browni (Dilkes, 1995; Dilkes, 1998). Howesia browni contrasts with the latter species in the presence of a medially expanded maxillary tooth plate, bearing multiple tooth rows, as occurs in rhynchosaurids (Dilkes, 1995). This species was described in detail by Dilkes (1995) and its tarsal anatomy was previously revised by Carroll (1976). The whereabouts of a pectoral girdle and a humerus described by Broom (1906) are currently unknown for SAM-PK-5885 (Dilkes, 1995), and the only available information for these elements is the original description of the species by Broom (1906).

Eohyosaurus wolvaardti Butler et al., 2015

Age. Early Anisian, early Middle Triassic (Rubidge, 2005).

Locality. Farm Lemoenfontein 44, Rouxville District, Free State Province, South Africa (Butler et al., 2015).

Stratigraphic horizon. Burgersdorp Formation, Cynognathus AZ subzone B, Tarkastad Subgroup, Beaufort Group, Karoo Supergroup, Karoo Basin (Butler et al., 2015).

Holotype. SAM-PK-K10159: partial skull missing the anterior end, with associated incomplete lower jaw and two partial limb bones, one of them preserved as a natural mould (Butler et al., 2015).

Diagnosis. Eohyosaurus wolvaardti is a small rhynchosaur that was diagnosed by Butler et al. (2015: 575) by the following autapomorphy: jugal with elongate dorsal process that forms the entire anterior margin of the infratemporal fenestra and that articulates anteriorly with the entire posterior margin of an elongate ventral process of the postorbital. In addition, Butler et al. (2015) distinguished this species from other rhynchosaurs on the basis of the following unique combination of character-states: maxillae and dentaries mediolaterally expanded; teeth present on the occlusal and lingual surfaces of the maxillae and dentaries; maxilla lacks a longitudinal occlusal groove and dentary lacks occlusal blade; occlusal margin of maxilla offset ventrally from the ventral margin of the main body of the jugal; presence of a short anguli oris crest on the lateral surface of the maxilla; posterior process of jugal is short and terminates at approximately 50% of the anteroposterior length of the infratemporal fenestra; elongate posterior process of the postorbital terminates above the anterior margin of the ventral process of the squamosal; elongate ventral process of the squamosal extends for more than 50% of the posterior margin of the infratemporal fenestra; and sagittal crest on parietal.

Remarks. Butler et al. (2015) named Eohyosaurus wolvaardti based on a single, partial skull and two partial limb bones (one preserved as a natural mould). This species was recovered as the sister-taxon of Rhynchosauridae (i.e., Rhynchosaurus articeps and more advanced rhynchosaurs), thus partially bridging the morphological gap between the other early rhynchosaurs from the Cynognathus AZ of South Africa and rhynchosaurids (Butler et al., 2015).

Rhynchosaurus articeps Owen, 1842

Age. Anisian, early Middle Triassic (Benton et al., 1994; Lucas, 2010).

Locality. Grinshill quarry, north of Shrewsbury, central England, UK (Benton, 1990; Benton & Spencer, 1995).

Stratigraphic horizon. Tarporley Siltstone Formation (Benton, 1990).

Lectotype. SHYMS 1: nearly complete skull and mandible.

Paralectotype. SHYMS 2: anterior part of a postcranium, including dorsal vertebrae and ribs, right scapula and coracoid, interclavicle, and right humerus.

Referred material. SHYMS 3: nearly complete skull, cervical and dorsal vertebrae and ribs, some gastralia, left coracoid, left forelimb, and partial left hindlimb; SHYMS 4: part and counterpart of a partial postcranium, including cervical and dorsal vertebrae and ribs, gastralia, right scapula and forelimb, part of the pelvic girdle and hindlimbs; SHYMS 5: fragments of dorsal vertebrae and ribs, gastralia, partial right pelvic girdle, fragments of left femur and complete right hindlimb; SHYMS 6: dorsal vertebrae and ribs, partial right pectoral girdle, right forelimb and possible skin impressions; SHYMS 7: partial right ischium, anterior caudal vertebrae and proximal end of right femur; SHYMS G3851: a sequence of anterior–middle caudal vertebrae and ribs; SHYMS G07537: an articulated series of posterior dorsal, sacral and anterior caudal vertebrae; NHMUK PV R1236: nearly complete skull and mandible; NHMUK PV R1237: partial skull and mandible that goes with the postcranium NHMUK PV R1238; NHMUK PV R1238: partial postcranium that misses left forelimb and hindlimb and tail; NHMUK PV R1239: part and counterpart that preserve impressions of the left hemimandible, gastralia, pectoral girdle, partial right forelimb, pelvic girdle and left hindlimb; NHMUK PV R1240: two blocks with caudal vertebrae that fits with NHMUK PV R1241; NHMUK PV R1241: hindlimb fragments; BRLSI M20a, b: two blocks with dorsal and caudal vertebrae, dorsal ribs, gastralia, partial pelvic girdle and right hindlimb; MANCH L7642: skull and cervical vertebrae; MANCH L7643: ribs; Keele University, unnumbered: skull and partial skeleton embedded within three blocks (unprepared) (modified from Benton, 1990).

Emended diagnosis. Small rhynchosaur that differs from other archosauromorphs in the following combination of features: laterally facing and sub-circular orbit with a raised and thickened border; anterior process of the jugal dorsoventrally narrow, being lower than the portion of maxilla located immediately below it; short and subtriangular posterior process of the jugal that does not reach the quadratojugal dorsal surface, resulting in an incomplete lower temporal bar; incipient anterior process of the quadratojugal; flat dorsal surface of the prefrontal; dorsal surface of the frontal with a deep, V-shaped depression on its posterior end; maxillary lateral tooth bearing area crest-shaped; posteromedially-to-anterolaterally oriented ridge on the posterior end of the palatal process of the pterygoid; posterior dorsal and sacral vertebrae with a deep fossa placed immediately lateral to the base of the neural spine; and second sacral rib bifurcated distally, with a tapering posterolateral process.

Remarks. Rhynchosaurus articeps was the first rhynchosaur genus and species to be erected (Owen, 1842), but it was not comprehensively described until the monographic work by Benton (1990). Rhynchosaurus articeps is currently the best-known Middle Triassic rhynchosaurid. Some reinterpretations of the anatomy of the species were recently made by Ezcurra, Montefeltro & Butler (2016), such as the presence of an open lower temporal bar and a distinctly bifurcated second sacral rib. This species has been frequently included as a representative of rhynchosaur anatomy in phylogenetic analyses focused on basal archosauromorphs (e.g., Dilkes, 1998; Gottmann-Quesada & Sander, 2009; Pritchard et al., 2015).

Bentonyx sidensis Langer et al., 2010a

Age. Late Anisian, Middle Triassic (Benton et al., 1994; Hounslow & McIntosh, 2003).

Locality. Pennington Point, 20 metres west of the Sid River outfall, Devon, southwest England, UK (Hone & Benton, 2008; Langer et al., 2010a).

Stratigraphic horizon. Otter Sandstone Formation (Hone & Benton, 2008; Langer et al., 2010a).

Holotype. BRSUG 27200: nearly complete skull, lacking the lower part and the posterolateral corners of the temporal areas, and partial mandible lacking most of the post-dentary bones (Langer et al., 2010a).

Diagnosis. Langer et al. (2010a: 1884) recognized two autapomorphies in their diagnosis of Bentonyx sidensis: a rounded depression on the ventral surface of the basisphenoid; and exceptionally large basal tubera. In addition, Langer et al. (2010a) listed the following features to distinguish Bentonyx sidensis from the likely sympatric species Fodonyx spenceri: narrower posterior margin of the skull (maximum width subequal to total skull length); a slender anterior process of the jugal (subequal in depth to the underlying portion of the maxilla); anterior margin of the quadrate process of the pterygoid forming an angle of less than 50°to the sagittal line; and maxillary tooth-bearing plates corresponding to more than half of the palatal length, measured from the anterior tip of the vomer to the posterior margin of the pterygoid (not including the posterior projection of the quadrate process).

Remarks. Hone & Benton (2008) described a new, fairly complete skull from the Otter Sandstone (BRSUG 27200) and referred it to Fodonyx spenceri. Subsequently, Langer et al. (2010a) revisited the taxonomy of the Devon rhynchosaur specimens and found strong evidence in support of a taxonomic distinction between the holotype of Fodonyx spenceri (EXEMS 60/1985.292) and the recently described skull (BRSUG 27200). The latter specimen was interpreted as the holotype of a new genus and species: Bentonyx sidensis. The holotype of Bentonyx sidensis represents one of the most exquisitely preserved Middle Triassic rhynchosaur skulls.

Eorasaurus olsoni Sennikov, 1997

Age. Late Capitanian–Wuchiapingian, midde–late Permian (Taylor et al., 2009; Ezcurra, Scheyer & Butler, 2014).

Locality. Right bank of the Volga River, village of Il’inskoe, Tetyushi District, European Russia (Sennikov, 1997).

Stratigraphic horizon. North Dvina Gorizont, Tarstan (Sennikov, 1997).

Holotype. PIN 156/109: two middle cervical vertebrae, one fairly complete and the second represented by the anterior half (Ezcurra, Scheyer & Butler, 2014).

Referred specimens. PIN 156/108: three posterior cervical vertebrae, the first of which is represented by the posterior end only; PIN 156/110: four partial anterior dorsal vertebrae and proximal portion of left anterior rib; PIN 156/111: two indeterminate long bones and some bone fragments (Ezcurra, Scheyer & Butler, 2014). The holotype and referred specimens are interpreted as belonging to the same individual (Sennikov, 1997; Ezcurra, Scheyer & Butler, 2014).

Emended diagnosis. Small archosauromorph that differs from other diapsids in the following combination of features (autapomorphies indicated with an asterisk): posterior cervical vertebrae with paradiapophyseal, posterior centrodiapophyseal and prezygodiapophyseal laminae; accessory, posterodorsally-to-anteroventrally oriented lamina partially subdividing the centrodiapophyseal fossa*; median keel on the ventral surface of the posterior cervical vertebrae; posterior cervical intercentra; long diapophysis in anterior dorsal vertebrae; and anterior dorsal rib with a thin lamina connecting the tuberculum and capitulum.

Remarks. Eorasaurus olsoni was originally erected and identified as a protorosaurid archosauromorph by Sennikov (1997). Although this species is one of the oldest known archosauromorphs, subsequent authors have ignored it. Ezcurra, Scheyer & Butler (2014) revisted the anatomy of Eorasaurus olsoni and reinterpreted some features, and included it for the first time in a quatitative phylogenetic analysis. Sennikov (1997) interpreted Eorasaurus olsoni as closely related to Protorosaurus speneri, but Ezcurra, Scheyer & Butler (2014) recovered this species as a possible member of Archosauriformes and Protorosaurus speneri as a basal archosauromorph. As a result, Eorasaurus olsoni is potentially the oldest known archosauriform (and also saurian) body fossil (Ezcurra, Scheyer & Butler, 2014; Bernardi et al., 2015).

Prolacertoides jimusarensis Young, 1973a

Age. Induan, Early Triassic (Young, 1973a; Lucas, 2010).

Locality. Dongxiaolongkou, Xinjiang Autonomous Region, China (Young, 1973a).

Stratigraphic horizon. Lower part of the Jiucaiyuan Formation, Cangfanggou Group (Young, 1973a).

Holotype. IVPP V3233: dorsoventrally compressed partial snout.

Emended diagnosis. Small archosauromorph that differs from other diapsids in the following combination of features: maxilla with a convex anterodorsal margin; 19 tooth positions in the maxilla; straight, conical and unserrated maxillary tooth crowns; well developed and subtriangular medial process on the prefrontal; and anterior processes of the pterygoids contact to each other extensively.

Remarks. Prolacertoides jimusarensis was briefly described by Young (1973a). The specimen is rather dorsoventrally compressed and, as a result, the lateral surfaces of the left maxilla and lacrimal face dorsolaterally (Fig. 7). Part of the possible left premaxilla is preserved in semiarticulation with the maxilla. Both maxillae, nasals, prefrontals, vomers, palatines, pterygoids, left ectopterygoid, partial right ectopterygoid, and probably the anterior ends of the jugals are preserved. The surfaces of the bones are generally well preserved, but some of them are severely damaged, such as the right maxilla and the anterior end of the left maxilla. The specimen is covered with glue and, as a consequence, sutures are usually difficult to discern.

Figure 7 Prolacertoides jimusarensis.

Holotype partial skull (IVPP V3233) in (A) dorsal; (B) ventral; (C) right lateral; and (D) left lateral views, and close up of right anterior maxillary tooth crowns. Numbers indicate character-states scored in the data matrix and the arrows indicate anterior direction. Abbreviations: bs, basisphenoid; ect, ectopterygoid; ju, jugal; la, lacrimal; lmtr, left maxillary tooth row; lprf, left prefrontal; mx, maxilla; na, nasal; pl-pt, palatine-pterygoid; rmtr, right maxillary tooth row; rprf, right prefrontal. Scale bars equal 5 mm, and 0.5 mm in the close up.

The snout is anteriorly tapering both in lateral and dorsal views. Indeed, the maxillae are strongly divergent from each other posteriorly, but this could be exaggerated by the post-mortem dorsoventral compression that the specimen has suffered. The posterior border of the external nares and the anterior half of the border of the orbit are preserved. An antorbital fenestra is absent. The anterior portion of the maxilla tapers anteriorly and its anterodorsal margin is slightly convex in lateral view, contrasting with the concave margin present in Jesairosaurus lehmani (ZAR 06) and some tanystropheids (e.g., Amotosaurus rotfeldensis: SMNS 90601; Tanystropheus longobardicus: Nosotti, 2007). As a result, it seems that the maxilla did not participate in the border of the external naris. The ascending process is anteroposteriorly broad and well developed dorsally, reaching the lateral margin of the skull roof. The suture with the lacrimal extends anterodorsally-to-posteroventrally and is posteriorly concave. It appears that the anterior process of the jugal overlaps laterally the posterior end of the maxilla, but this interpretation should be treated with caution. The anterior tip of the jugal apparently extended anteriorly beyond the level of the anterior border of the orbit, contrasting with the condition in several basal saurians (e.g., Jesairosaurus lehmani: ZAR 06; Gephyrosaurus bridensis: Evans, 1980). The maxillary tooth row finishes well anterior to the level of the anterior border of the orbit, contrasting with the more posteriorly extending tooth row present in other basal saurian (e.g., Gephyrosaurus bridensis: Evans, 1980; Protorosaurus speneri: Gottmann-Quesada & Sander, 2009; Jesairosaurus lehmani: ZAR 07; Macrocnemus bassanii: PIMUZ T4822). There are approximately 19 tooth positions based on the right maxilla, but the tooth count in the left maxilla cannot be determined because its posterior end is still covered with matrix. The maxillary tooth crowns are straight in labial view and lack denticles. In cross-section the anterior and middle maxillary crowns are circular and the posterior crowns are circular to slightly labiolingually compressed. The morphology of the teeth resembles that of Jesairosaurus lehmani (ZAR 06, 07), Amotosaurus rotfeldensis (SMNS 90601) and Youginia capensis (SAM-PK-K6205), but contrasts with the distally curved crowns of Macrocnemus bassanii (PIMUZ T4822) and Prolacerta broomi (BP/1/471, 4504a). Young (1973a) described and figured a middle maxillary tooth with a mesiodistal constriction at the base of the crown. Unfortunately, this tooth is now broken off and this feature could not be checked. Nevertheless, the first maxillary tooth crown is fairly complete and lacks a constriction at its base. There is no evidence of ankylosis between the teeth and the maxilla.

The anterior end of the nasals is damaged, but it seems that the posterior border of the external nares is preserved (cf. Young, 1973a). It seems that the external nares are marginal and extended posteriorly to the level of the suture between the premaxilla and maxilla. The external naris does not taper posterodorsally, contrasting with some basal lepidosauromorphs (e.g., Gephyrosaurus bridensis: Evans, 1980). The nasal is laterally expanded at its posterior end and possesses a long, anterolaterally oriented suture with the prefontal. By contrast, the suture between the prefrontal and nasal is mainly anteroposteriorly oriented in Prolacerta broomi (BP/1/471) and archosauriforms (e.g., Proterosuchus fergusi: BSPG 1934 VIII 514, RC 846, SAM-PK-K10603; Erythrosuchus africanus: BP/1/5207, NMQR 1473). The lacrimal is a slit-like, anterodorsally-to-posteroventrally oriented bone that resembles in morphology that of Prolacerta broomi (BP/1/471). There is no anterior process of the lacrimal and the bone possibly formed part of the anterior border of the orbit, but the latter condition cannot be determined confidently. The lacrimal is relatively anteroposteriorly broad and, as a result, the bone is well exposed on the lateral surface of the snout. The anterior process of the jugal seems to have been anteroposteriorly long and forms the ventral border of the orbit. The prefontals are widely exposed on the skull roof at level with the anterior border of the orbits, resembling the condition in Prolacerta broomi (Modesto & Sues, 2004). The prefrontal possesses a subtriangular medial projection that strongly constrictes transversely the suture between the nasals and frontals, as occurs in Trilophosaurus buettneri (Spielmann et al., 2008) and Gephyrosaurus bridensis (Evans, 1980). The dorsal surface of the prefrontal is smooth and transversely convex, lacking any change in slope between the dorsal and lateral surfaces of the bone. The prefrontal forms the anterodorsal border of the orbit, but the ventral process is not preserved on both sides and, as a result, it is not possible to determine how ventrally it extended. There is a circular, large foramen on the lateral surface of the prefrontal, immediately dorsal to the suture with the lacrimal and adjacent to the anterior border of the orbit. This foramen may represent the exit of the naso-lacrimal duct.

The pterygoids contact each other extensively at the median line, contrasting with the condition in Prolacerta broomi (Modesto & Sues, 2004) and basal archosauriforms (e.g., Proterosuchus fergusi: RC 59; Proterosuchus goweri: NM QR 880), in which there is an interpterygoid vacuity. It is not possible to discern the suture between vomers and pterygoids, nor between palatines and pterygoids. The choanae cannot be recognized, but they should have been placed anteriorly in the palate. The pterygoids-palatines are strongly expanded transversely and anteroposteriorly in the palate. The transverse process of the pterygoid possesses a posterolaterally oriented posterior margin. The left ectopterygoid is well preserved and was labelled as pterygoid by Young (1973a: Fig. 1). The ectopterygoid expands anteroposteriorly at its lateral end, acquiring a fan-shaped morphology in ventral view, closely resembling the morphology present in Gephyrosaurus bridensis (Evans, 1980). The ectopterygoid articulates laterally only with the jugal. It was not possible to recognize any palatal teeth in the pterygoids and palatines, but this could be a result of poor preservation. The quadrate process of the pterygoid is posterolaterally oriented and long, but the contact with the quadrate is not preserved or discernable. The suborbital fenestra is antroposteriorly very short, contrasting with the condition usually present in basal archosauriforms. Between the quadrate processes of the pterygoids there is a bone with a strongly concave ventral surface that likely represents the parasphenoid or parabasisphenoid (cf. Young, 1973a).

Prolacerta broomi Parrington, 1935

Age. Induan, earliest Triassic (Groenewald & Kitching, 1995).

Localities. Harrismith Commonage, Harrismith District, Free State, South Africa (type locality; Parrington, 1935). Referred specimens were collected from several localities in the Katberg Formation (Lystrosaurus AZ), South Africa (see Gow, 1975; Modesto & Sues, 2004).

Stratigraphic horizon. Katberg Formation, Lystrosaurus AZ, Tarkastad Subgroup, Beaufort Group, Karoo Supergroup, Karoo Basin (Parrington, 1935; Gow, 1975; Modesto & Sues, 2004). In addition, Colbert (1987) referred specimens from the Fremouw Formation (earliest Triassic) of the Transantarctic Mountains of Antarctica to Prolacerta broomi (see below).

Holotype. UMZC 2003.40: a partial skull and mandible.

Referred material. AMNH 9502: postcranial skeleton; BP/1/471: complete skull with attached mandible; BP/1/2675: nearly complete skull, now mostly disarticulated, with postcranial skeleton; BP/1/2676: nearly complete skeleton; BP/1/4504a: skull of a small individual; BP/1/5066: partial, flattened skull; BP/1/5375: skull, complete from mid-snout to occiput with partial mandible; GHG 431: partial skull and mandible, lacking the snout; SAM-PK-K10018: fairly complete, somewhat weathered skull and mandible; SAM-PK-K10797: well-preserved skull and posterior half of mandible attached to anterior cervical vertebrae; UCMP 37151: complete skull with articulated cervical vertebrae.

Emended diagnosis. Modesto & Sues (2004: 336) provided a diagnosis of Prolacerta broomi, which is modified here as follows. Prolacerta broomi is a basal archosauromorph distinguished from other saurians by the presence of: septomaxillae; notch on the ventral margin of the alveolar margin along the premaxilla–maxilla suture; conspicuous posterolateral exposure of the lacrimal duct openings; well developed posterolateral process on the frontal, resulting in an acute-angled and V-shaped suture between frontals and parietals; parietals lacking a sub-rectangular fossa on the posterior half of the dorsal surface of the bones; absence of postparietals; extensive contact between the surangular and the prearticular in the articular region of the lower jaw.

Remarks. Parrington (1935) named Prolacerta broomi and considered it as a thecodont intermediate between basal diapsids and lizards. Subsequently, Camp (1945) described a new specimen and interpreted it as the closest relative of the Permian Protorosaurus speneri, thus germinating the concept of a monophyletic “Protorosauria.” More specimens of Prolacerta broomi were collected from the Lystrosaurus AZ of South Africa during the subsequent 30 years and Gow (1975) provided for the first time a rather comprehensive description of the anatomy of the species. Colbert (1987) reported the presence of Prolacerta broomi from earliest Triassic beds of Antarctica, but these specimens are not considered for the scorings of the present analysis because a detailed, updated taxonomic revision of this material is needed. Modesto & Sues (2004) redescribed in detail the cranial anatomy of Prolacerta broomi based on both historical and recently collected specimens. Prolacerta broomi has been considered a key taxon in phylogenetic analyses, mainly to optimize the ancestral character-states of Archosauriformes. There is currently a general consensus that Prolacerta broomi represents the sister-taxon or one of the most closely related taxa of Archosauriformes (e.g., Dilkes, 1998; Modesto & Sues, 2004; Gottmann-Quesada & Sander, 2009; Nesbitt, 2011; Ezcurra, Scheyer & Butler, 2014; Pritchard et al., 2015), contrasting with early cladistics analyses that found this species as more closely related to Protorosaurus speneri and tanystropheids (e.g., Benton, 1985; Jalil, 1997) than to archosauriforms.

First-hand study of most available specimens of Prolacerta broomi allowed the reinterpretion of some anatomical features that do not agree with the skull reconstruction recently published by Nesbitt (2011). Since some of these features are of potential phylogenetic significance, they are discussed as follows. The anterior tip of the maxilla of the referred specimen of Prolacerta broomi BP/1/471 possesses an anteriorly opening notch, which is placed immediately below the articulation between the maxilla and the postnarial process of the premaxilla. This condition is very similar to that present in proterosuchids (e.g., RC 846) and is identified as an interruption of the alveolar margin of the skull at the level of the premaxilla–maxilla suture, which was not described or illustrated by previous authors (Gow, 1975; Modesto & Sues, 2004; Nesbitt, 2011). An interrupted alveolar margin between the premaxilla and maxilla is also preserved in the referred specimen of Prolacerta broomi BP/1/4504a. Nesbitt (2011: Fig. 16A) reconstructed the jugal of Prolacerta broomi as completely lacking a posterior process. However, this process was described and illustrated by previous authors (Gow, 1975; Modesto & Sues, 2004) and its presence is supported here based on multiple specimens (e.g., BP/1/2675, 5375; SAM-PK-K10797). The quadratojugal has been reconstructed by Gow (1975) as a slightly ventrally extended, splint-like bone that does not reach the distal end of the quadrate. However, the quadratojugal reaches and partially overlap the ventral end of the quadrate in lateral view, with a moderately posteriorly expanded ventral end, in at least one well-preserved referred specimen (SAM-PK-K10797). The retroarticular process of the lower jaw is strongly bowed dorsally, resembling the condition in Proterosuchus alexanderi (NMQR 1484), but contrasting with previous reconstructions (e.g., Gow, 1975; Nesbitt, 2011).

Kadimakara australiensis Bartholomai, 1979

Age. Induan, earliest Triassic (Warren & Hutchinson, 1990; Warren, Damiani & Yates, 2006).

Locality. 72 kilometres southwest of Rolleston, central Queensland, Australia (Bartholomai, 1979).

Stratigraphic horizon. Lower beds of the upper part of the Arcadia Formation, Rewan Group (Bartholomai, 1979).

Holotype. QMF 6710: temporal region of skull and right post-dentary bone of the lower jaw.

Referred material. QMF 6676: partial snout and anterior half of the lower jaw.

Emended diagnosis. Kadimakara australiensis is a basal archosauromorph distinguished from other saurians by the presence of (autapomorphy indicated with an asterisk): well developed posterolateral process on the frontal, resulting in an acute-angled and V-shaped suture between frontals and parietals; parietals with a sub-rectangular fossa on the posterior half of the dorsal surface of the bones*; and absence of postparietals. Although Bartholomai (1979: 226) included in his original diagnosis of the species the autapomorphy enumerated here for the species, an emended diagnosis was provided here because the putative postorbital and postfrontal described by Bartholomai (1979) are reinterpreted here as squamosal, and postorbital and postfrontal, respectively (see below).

Remarks. Bartholomai (1979) named the new genus and species Kadimakara australiensis and recognized its close similarities with the South African species Prolacerta broomi. Subsequent authors proposed that Kadimakara australiensis is possibly a subjective junior synonym of Prolacerta broomi (Borsuk-Białynicka & Evans, 2009; Evans & Jones, 2010). However, this hypothesis of synonymy is not followed here because Kadimakara australiensis possesses a median subrectangular fossa on the posterior half of the parietals that is separated from the margins of the supratemporal fossae by broad flat surfaces (QM F6710) (Fig. 8: mfo). This condition resembles in its position the pineal fossa of proterosuchids (e.g., Proterosuchus fergusi: TM 201, SAM-PK-K10603; Proterosuchus goweri: NMQR 880; Proterosuchus alexanderi: NMQR 1484) and some erythrosuchids (e.g., Erythrosuchus africanus: BP/1/5207, NHMUK PV R3592, NMQR 1473). By contrast, all studied specimens of Prolacerta broomi lack a fossa on the posterior half of the parietals or possesses a median fossa that is confluent with the margins of the supratemporal fossae (BP/1/471, 2675, 4504a, 5375; GHG 431; SAM-PK-K10797; UMCZ 2003.41R). As a result, the morphology of the dorsal surface of the parietals allows these closely related species to be distinguished from each other.

Figure 8 Kadimakara australiensis.

(A, B) Holotype postorbital region of skull (QMF 6710) and (C, D) referred snout (QMF 6676) in (A) dorsal; (B, C) right lateral; (D) and left lateral views. Numbers indicate character-states scored in the data matrix and the arrows indicate anterior direction. Abbreviations: apsq, anterior process of the squamosal; dt, dentary; fr, frontal; la, lacrimal; mfo, median fossa; mpsq, medial process of the squamosal; mx, maxilla; po, postorbital; pofr, postfrontal; ppsq, posterior process of the squamosal; qj, quadratojugal; sa, surangular; stf, supratemporal fenestra; stfo, supratemporal fossa; vpsq, ventral process of the squamosal. Scale bars equal 2 mm.

The first-hand study of the hypodigm of Kadimakara australiensis allowed the reinterpretion of some of the bones originally identified by Bartholomai (1979). This author described the postorbital as having a long, deep and anteriorly directed medial process and that the sutural relationship with the squamosal is difficult to interpret because it appears to be very close. The reinterpretion presented here is that the element identified by Bartholomai (1979) mainly as a postorbital is actually a squamosal and only the distal end of the ascending process of the postorbital is preserved (Fig. 8). The reinterpreted squamosal possesses a morphology that closely resembles that of Prolacerta broomi, with an anteroventrally oriented, straight and subrectangular ventral process that forms with the anterior process a squared posterodorsal corner of the infratemporal fenestra (QMF 6710). The preserved distal end of the ascending process of the postorbital is medially oriented and forms the anterior border of the supratemporal fenestra, excluding the postfrontal from the border of this opening.

The holotype of Kadimakara australiensis is based on the postorbital region of a skull (QMF 6710). A partial snout (QMF 6676) from the same locality was referred to the species, but with no clear association (Bartholomai, 1979). There are no overlapping bones between the specimens, and, as a result, their assignment to the same species is ambiguous. Nevertheless, the morphologies of both specimens are consistent with that of an animal similar to the non-archosauriform archosauromorph Prolacerta broomi. As such, the referral of QMF 6676 to Kadimakara australiensis can be considered a working hypothesis until there is more complete information that would allow it to be supported or rejected. Therefore, the holotype and complete hypodigm (holotype + referred specimen) of Kadimakara austaliensis were included as different terminals.

Boreopricea funerea Tatarinov, 1978

Age. Olenekian, late Early Triassic (Sennikov, 2008).

Occurrence. Core sample at 1,112.3 metres depth, Kolguyev Island, Arkhangel Province, Arctic Russia (Tatarinov, 1978; Benton & Allen, 1997).

Holotype. PIN 3708/1: nearly complete skull and postcranium, lacking the pelvis, posterior dorsal and anterior caudal vertebrae (Benton & Allen, 1997).

Referred material. PIN 3708/2: anterior end of the snout (Tatarinov, 1978). This specimen could not be located by Benton & Allen (1997).

Emended diagnosis. Benton & Allen (1997: 932) provided an emended diagnosis for Boreopricea funerea, which is modified here based on some anatomical reinterpretations of the specimen. Boreopricea funerea is a basal archosauromorph distinguished from other saurians by the presence of: jugal-squamosal contact; straight fronto-parietal suture; extremely reduced or absence of posterior process of jugal; more than seven cervical vertebrae; mammillary processes on the neural spines of cervico-dorsal vertebrae; metacarpal III equal in length to, or longer than, metacarpal IV; and second phalanx on pedal digit V is long compared to other phalanges.

Remarks. Tatarinov (1978) named the new Triassic genus and species Boreopricea funerea and considered that it was an intermediate form between Prolacerta broomi from South Africa and Macrocnemus bassanii from western Europe. The anatomy of the species was reviewed by Benton & Allen (1997) and they mentioned that the specimen had been considerably damaged since its original description. These authors agree in the general similarity between Boreopricea funerea and Prolacerta broomi. Benton & Allen (1997) included Boreopricea funerea in a quantitative phylogenetic analysis for the first time, which was focused almost exclusively in prolacertiforms, and recovered it as more closely related to tanystropheids than to Prolacerta broomi and Protorosaurus speneri. A very similar result was found independently by Jalil (1997). Boreopricea funerea was scored based on first-hand observations of the specimen and complemented with the publications of Tatarinov (1978) and Benton & Allen (1997). The interpretation of some anatomical features from the skull differs between both authors and cannot be evaluated due to the very bad current state of preservation of this anatomical region (PIN 3708/1). These characters were scored as missing data in the current data matrix.

Archosaurus rossicus Tatarinov, 1960

Age. Late Changhsingian, latest Permian (Rubidge, 2005; Sennikov & Golubev, 2006; Sennikov & Golubev, 2012; Benton, Tverdokhlebov & Surkov, 2004; Krassilov & Karasev, 2009).

Locality. Vyazniki locality, near Vyazniki, Vladamir Province, Russia (Tatarinov, 1960; Sennikov, 1988a).

Stratigraphic horizon. Uppermost part of the Tatarian series, “Vyazniki Biotic Assemblage” (Tatarinov, 1960; Sennikov, 1988a).

Holotype. PIN 1100/55: isolated left premaxilla lacking the distal portion of the prenarial and postnarial processes and teeth.

Putative referred material. PIN 1100/78: left dentary; PIN 1100/48: skull roof; PIN 1100/85: four tooth crowns; PIN 1100/66, 66a, 66b: three cervical vertebrae. Tatarinov (1960) and Sennikov (1988a) also referred other cranial and postcranial bones that are not included here among the putative referred material of the species because their morphology is not consistent with that of a proterosuchid (e.g., squamosal reinterpreted as an indeterminate bone by Ezcurra, Scheyer & Butler (2014)) or also resemble that of other archosauromorph lineages and lack archosauriform apomorphies (e.g., proximal end of tibia) (Ezcurra, Scheyer & Butler, 2014).

Emended diagnosis. Archosaurus rossicus is a proterosuchid distinguished from other diapsids on the basis of the following unique combination of characters-states: main axis of the palatal process of the premaxilla forming an angle of about 20°with respect to the main axis of the postnarial process, probably indicating the presence of a strongly downturned premaxilla; more than five tooth positions in the premaxilla; first four premaxillary alveoli open lateroventrally; and strongly acute angle formed between the anterior margin of the premaxillary body and the alveolar margin. The premaxilla of Archosaurus rossicus is extremely similar to that of Proterosuchus fergusi, Proterosuchus goweri and “Chasmatosaurus” yuani, but it differs in the last feature listed in the diagnosis.

Remarks. Archosaurus rossicus is the oldest known unequivocal archosauriform (Tatarinov, 1960; Charig & Reig, 1970; Charig & Sues, 1976; Sennikov, 1988a; Gower & Sennikov, 2000; Nesbitt, 2011; Ezcurra, Butler & Gower, 2013; Ezcurra, Scheyer & Butler, 2014), but it has been included only recently in quantitative phylogenetic analyses (Nesbitt, 2011; Ezcurra, Scheyer & Butler, 2014). These analyses recovered Archosaurus rossicus as a proterosuchid, as proposed by previous authors. Nesbitt (2011) and Ezcurra, Scheyer & Butler (2014) discussed the assignment of the specimens previously referred to Archosaurus rossicus and considered these referrals problematic because of the lack of overlapping characters with the holotype and the fact that the specimens come from different stratigraphic levels within a geographically large locality with a stratigraphic thickness of around 25 metres (Ezcurra, Scheyer & Butler, 2014). As a result, Nesbitt (2011) and Ezcurra, Scheyer & Butler (2014) restricted the scorings of Archosaurus rossicus to its holotype premaxilla, and this decision is also followed here. The hypodigm of Archosaurus rossicus was described by Tatarinov (1960) and Sennikov (1988a), and the anatomical knowledge of the holotype was recently complemented by Ezcurra, Scheyer & Butler (2014).

Holotype of Proterosuchus fergusi

Age. Induan, earliest Triassic (Rubidge, 2005; Lucas, 2010).

Locality. Farm Wheatlands, Tarkastad, Chris Hani District, Eastern Cape Province, South Africa (Broom, 1903a).

Stratigraphic horizon. Upper Balfour Formation or lower Katberg Formation, Lystrosaurus AZ, Beaufort Group, Karoo Supergroup (Broom, 1903a).

Holotype. SAM-PK-591: partial, poorly preserved skull and lower jaw (Broom, 1903a; Welman, 1998; Ezcurra & Butler, 2015a).

Remarks. Proterosuchus fergusi has a long and conflicting taxonomic history that has been recently summarized by Ezcurra & Butler (2015a). Broom (1903a) erected this genus and species on the basis of the first proterosuchid specimen collected from the Early Triassic of South Africa: a poorly preserved partial skull and lower jaw (SAM-PK-591) that was diagnostic at that time. Subsequently, Hughes (1963) considered Proterosuchus fergusi an invalid archosauriform species, but this interpretation was not followed by subsequent authors (e.g., Hoffman, 1965; Charig & Reig, 1970; Cruickshank, 1972; Charig & Sues, 1976; Welman, 1998). Welman (1998) considered that all the proterosuchid species described at that time from the Lystrosaurus AZ of South Africa were subjective junior synonyms of Proterosuchus fergusi and proposed a revised diagnosis for the genus and species. However, Ezcurra & Butler (2015a) revised the taxonomy of the proterosuchid specimens from South Africa, in which they redescribed in detail the holotype of Proterosuchus fergusi and argued that it cannot be distinguished from other proterosuchids (e.g., “Chasmatosaurus” yuani from China). As a result, following the recommendations of the International Code of Zoological Nomenclature, Ezcurra & Butler (2015a) proposed a neotype (RC 846) for Proterosuchus fergusi. The South African proterosuchid species “Chasmatosaurus vanhoepeni” and “Elaphrosuchus rubidgei” were considered junior synonyms of Proterosuchus fergusi by Ezcurra & Butler (2015a). I decided here to score the holotype of Proterosuchus fergusi alone in order to test if the specimen represents a member of Proterosuchidae.

Proterosuchus fergusi Broom, 1903a

Age. Induan, earliest Triassic (Rubidge, 2005; Lucas, 2010).

Localities. Farm Ruygte Valley 321, Middelburg, Chris Hani District, Eastern Cape Province, South Africa (neotype locality). Referred specimens have been collected in multiple localities from the same horizon as the neotype locality in South Africa (see Ezcurra & Butler, 2015a: Table 3).

Stratigraphic horizon. Upper Balfour Formation and/or lower Katberg Formation, Lystrosaurus AZ, Beaufort Group, Karoo Supergroup (see Ezcurra & Butler (2015a) and references herein).

Proposed neotype. RC 846 (mistakenly referred to as RC 96 by Welman & Flemming (1993) and subsequent authors), large fairly complete skull and lower jaw, atlas, axis, partial third cervical vertebra and first three cervical ribs.

Referred material. BP/1/3993: medium-sized (38.8 cm total skull length) partial skull and lower jaw (lacking right temporal region and the posterior ends of the mandibular rami), axis and five anterior–middle postaxial cervical vertebrae; BP/1/4016: small (c. 24 cm total skull length) partial skull and lower jaw (lacking the anterior ends of the premaxillae and the anterior half of skull roof), first four cervical vertebrae and probable atlantal rib and postaxial cervical ribs; BP/1/4224: small (c. 23 cm total skull length) posterior half of skull and lower jaw, axis and one cervical rib; BSPG 1934 VIII 514: large (43.5 cm total skull length) partial skull and complete lower jaw, first four cervical vertebrae with their ribs and intercentra; GHG 231: large (47.7 cm total skull length) partial skull (lacking the left maxilla) and complete lower jaw, first seven cervical vertebrae, and atlantal, axial, fourth and fifth cervical ribs; GHG 363: large partial snout, three cervicodorsal vertebrae with two right ribs in partial articulation, two middle dorsal vertebrae, one middle caudal vertebra, one posterior caudal vertebra, and interclavicle; RC 59 (holotype of “Elaphrosuchus rubidgei”): small (17.8 cm total skull length) partial skull and lower jaw (lacking prefrontals, lacrimals, left squamosal and quadratojugal, epipterygoids and braincase), an atlantal neural arch and some cervical ribs; SAM-PK-11208: medium-sized (35.0 cm total skull length) partial skull and lower jaw (lacking most of the skull roof and with the left side severely damaged), axis, third and fourth cervical vertebrae in articulation, probable fifth cervical to first dorsal vertebrae in articulation, three anterior dorsal vertebrae in articulation, possible first sacral vertebra and some long bone fragments; SAM-PK-K140: small (28.7 cm total skull length) partial skull (lacking the skull roof and braincase) and lower jaw, first four cervical vertebrae in articulation, a series of seven middle cervical to anterior dorsal vertebrae, two middle dorsal vertebrae, sacral vertebrae (now lost), several cervical and dorsal ribs and gastralia, right scapula (now lost), left ulna, radius, carpus and hand, partial pelvic girdle (currently lost), partial hindlimbs, including well-preserved left femur, right astragalus and calcaneum, and left foot in articulation and a bone previously identified as an isolated osteoderm (now only preserved as a mould); SAM-PK-K10603: large (c. 41 cm total skull length) fairly complete skull and lower jaw (missing most of the premaxillae) and atlas; and TM 201 (holotype of “Chasmatosaurus vanhoepeni”): large (c. 44 cm total skull length) partial skull and lower jaws. Botha-Brink, Huttenlocker & Modesto (2014: 300) briefly reported, but not figured, a new specimen (NMQR 3924) that was assigned to Proterosuchus fergusi on the basis of ‘the skull roof and tooth morphology’. This specimen was not discussed by Ezcurra & Butler (2015a) and in this contribution because it was not examined by the author.

Diagnosis. Proterosuchid archosauriform (skull length reaching up to approximately 50 cm and total body length up to 3–3.5 m) distinguished from other archosauromorphs on the basis of the following unique combination of character-states: premaxilla lacking a groove on the lateral surface of the main body; ratio of total length of maxilla versus length of maxilla anterior to the antorbital fenestra greater than 2.5; maxilla lacks an anterolaterally opening longitudinal groove adjacent to the anterior margin of the bone; minimum height of the horizontal process of maxilla is equal to or less than 13% of the total length of the maxilla; maxillary alveolar margin straight to gently convex in lateral view; quadrate with an angle between the posterior margins of the dorsal and ventral ends of less than 130°; presacral vertebrae with mammillary processes on the neural spines of at least cervicals 6 and 7 and absent on dorsals 4–7; and presence of postaxial intercentra (Ezcurra & Butler, 2015a: 164).

Remarks. See comments for the holotype of Proterosuchus fergusi. Different specimens currently referred to Proterosuchus fergusi have been described by several authors (e.g., Broom, 1903a; Broom, 1932; Broom, 1946; Haughton, 1924; Broili & Schröder, 1934; Gow, 1975) and the most comprehensive anatomical description of the species was conducted by Cruickshank (1972) and complemented by Welman (1998), Ezcurra & Butler (2015a) and Ezcurra & Butler (2015b). Nevertheless, a detailed and complete osteological description of the species is still unavailable and now in preparation by the author.

Proterosuchus goweri Ezcurra & Butler, 2015a

Age. Induan, earliest Triassic (Rubidge, 2005; Lucas, 2010).

Locality. Farm Kruisvlei (Kruisvlei 1095 in Brink (1955) and Kruisvlei 279 in Welman (1998)), east of Winburg, Lejweleputswa District, Free State Province, South Africa (Brink, 1955).

Stratigraphic horizon. Upper Balfour Formation or lower Katberg Formation, Lystrosaurus AZ, Beaufort Group, Karoo Supergroup (Brink, 1955).

Holotype. NMQR 880: partial skull with detached braincase, a right dorsal rib, and left tibia and fibula.

Diagnosis. Proterosuchid archosauriform (skull length of the only known individual c. 39 cm and total body length estimated in 2.5–3 m) distinguished from other archosauromorphs by the following combination of character-states (autapomorphies marked with an asterisk): premaxilla lacking grooves on the lateral surface; ratio of total length of maxilla versus length of maxilla anterior to the antorbital fenestra greater than 2.5; maxilla with an edentulous anterior end, the length of which is equivalent to that of two tooth positions; maxilla lacking an anterolaterally opening longitudinal groove adjacent to the anterior margin; minimum height of the horizontal process of maxilla is equal to or greater than 15% of the total length of the maxilla*; maxillary alveolar margin distinctly sigmoid in lateral or medial views, with a concave anterior two-thirds and a convex posterior third*; and quadrate with an angle between the posterior margins of the dorsal and ventral ends greater than 145°(Ezcurra & Butler, 2015a: 166, 167).

Remarks. Brink (1955) reported a new proterosuchid specimen (NMQR 880) from the Lystrosaurus AZ of South Africa that he assigned to “Chasmatosaurus vanhoepeni”. Welman (1998) considered all the proterosuchid specimens from South Africa referable to Proterosuchus fergusi, including NMQR 880. Ezcurra & Butler (2015a) noted that NMQR 880 possesses a unique combination of features that distinguish it from other archosauromorphs and used this specimen to erect the new species Proterosuchus goweri. The holotype of Proterosuchus goweri was briefly described by Brink (1955), and recently some anatomical comments were added by Ezcurra & Butler (2015a). A detailed description of the species is currently in preparation by the author.

Proterosuchus alexanderi (Hoffman, 1965)

Age. Induan, earliest Triassic (Rubidge, 2005; Lucas, 2010).

Locality. Farm Zeekoegat, four miles from Venterstad, Joe Gqabi District, Eastern Cape Province, South Africa (Hoffman, 1965).

Stratigraphic horizon. Upper Balfour Formation or lower Katberg Formation, Lystrosaurus AZ, Beaufort Group, Karoo Supergroup (Hoffman, 1965).

Holotype. NMQR 1484: small, fairly complete skull (lacking most of the premaxillae) and postcranial axial skeleton (lacking the posterior half of the caudal series) and partial appendicular skeleton.

Diagnosis. Proterosuchid archosauriform distinguished from other archosauromorphs by the following combination of character-states (autapomorphies marked with an asterisk): ratio of total length of maxilla versus length of maxilla anterior to the antorbital fenestra less than 2.3*; minimum height of the horizontal process of maxilla is equal to or less than 13% of the total length of the maxilla; maxillary alveolar margin straight to gently convex in lateral view; frontals with a dorsal surface ornamented by a series of anastomosed shallow grooves and subcircular pits with a seemingly random arrangement* (Ezcurra & Butler, 2015a: Figs. 4A and 4B); quadrate with an angle between the posterior margins of the dorsal and ventral ends greater than 145 degree; presacral vertebrae with mammillary processes on the neural spines present in presacrals 14–16 (dorsals 5–7)*; (7) presence of postaxial intercentra; and first sacral rib distally subdivided* (Ezcurra & Butler, 2015a: 166).

Remarks. “Chasmatosaurus” alexanderi was erected by Hoffman (1965) on the basis of a specimen that represents the most complete proterosuchid skeleton collected so far from the Lystrosaurus AZ of South Africa. This species was considered a subjective junior synonym of “Chasmatosaurus vanhoepeni” by later authors (e.g., Charig & Reig, 1970; Cruickshank, 1972; Charig & Sues, 1976; Welman, 1998), but Ezcurra & Butler (2015a) resurrected the species because it possessed a unique combination of character-states and autapomorphies that distinguish it from other proterosuchids. Since “Chasmatosaurus vanhoepeni” was considered a subjective junior synonym of Proterosuchus fergusi by Ezcurra & Butler (2015a), these authors tranfered “Chasmatosaurus” alexanderi to the genus Proterosuchus, resulting in the new combination Proterosuchus alexanderi. The holotype of Proterosuchus alexanderi was originally briefly described by Hoffman (1965) and described more in detail by Cruickshank (1972). Clark et al. (1993), Welman (1998), Klembara & Welman (2009) and Ezcurra & Butler (2015a) provided additional anatomical observations.

“Chasmatosaurus” yuani Young, 1936

Age. Induan, earliest Triassic (Rubidge, 2005; Lucas, 2010).

Localities. Shoukou Fukanghsien, Fuyuan County, Heilongjiang Province, People’s Republic of China (type locality). Referred specimens come from Hungshantig, Hotung, Heilongjiang Province, People’s Republic of China, and the same stratigraphic horizon as the holotype (Young, 1936; Young, 1963).

Stratigraphic horizon. Jiucaiyuan Formation, Lystrosaurus AZ (Young, 1936; Young, 1963).

Holotype. IVPP V36315: fragmentary skeleton composed of a partial snout and postcranium. The postcranium included axial and appendicular bones (see Young, 1936), but they could not be located in the collection of the IVPP (May 2013) and should be considered currently lost.

Referred material. IVPP V2719: partial braincase and postcranium, including elements of the axial series, pectoral girdle, forelimb, pelvic girdle and hindlimb; IVPP V4067: fairly complete articulated skeleton, missing the distal two-thirds of the tail, gastralia, right hand and left himdlimb stilopodium and autopodium.

Emended diagnosis. “Chasmatosaurus” yuani is a medium-sized basal archosauriform that differs from other basal archosauromorphs by the following combination of features (autapomorphies indicated with an asterisk): lateral surface of the premaxillary body with a pair of grooves originated dorsally from a circular neurovascular foramen situated on the anterior half of the premaxillary body*; anterior process of the maxilla with a distinct shelf extending posterodorsally from the anteroventral tip of the bone and restricted to the anterior third of the process*; and anterior tip of the maxilla edentulous and slightly ventrally oriented.

Remarks. See Nesbitt et al. (2015).

“Chasmatosaurus ultimus” Young, 1964 nomen dubium

Age. Anisian, early Middle Triassic (Liu, Li & Li, 2013).

Locality. Louzeyu Village, Wuhsiang County, Shanxi Province, People’s Republic of China (Young, 1958; Liu et al., 2015).

Stratigraphic horizon. Upper member of the Ermaying Formation (Young, 1958; Liu et al., 2015).

Holotype. IVPP V2301: incomplete anterior part of skull with lower jaw.

Remarks. IVPP V2301was originally described by Young (1958) as a referred specimen of “Chasmatosaurus” yuani, but subsequently Young (1964) erected the new species “Chasmatosaurus ultimus” on the basis on this specimen. This species has been considered the youngest known member of Proterosuchidae until recently (Ezcurra, Butler & Gower, 2013). The original diagnosis of “Chasmatosaurus ultimus” differentiated this species from “Chasmatosaurus” yuani on the basis of its smaller size and larger, more strongly recurved tooth crowns (Young, 1964). Liu et al. (2015) revised the specimen and concluded that it represents an indeterminate archosaur and the species is a nomen dubium. Although this species is invalid, it was included in the taxonomic sample of the present phylogenetic analysis to test the hypothesis that “Chasmatosaurus ultimus” is an archosaur, which was proposed by Liu et al. (2015) based on qualitative grounds.

“Ankistrodon indicus” Huxley, 1865 nomen dubium

Age. Induan, earliest Triassic (Rubidge, 2005; Lucas, 2010).

Locality. Deoli locality, close to Deoli village, West Bengal Province, India (Blanford in Huxley, 1865).

Stratigraphic horizon. Panchet Formation, Damodar Basin (Blanford in Huxley, 1865).

Holotype. GSI 2259: portion of distal end of horizontal process of the right maxilla with two teeth.

Remarks. The holotype of “Ankistrodon indicus” is based on an isolated maxillary fragment with two teeth. This species was the first proterosuchid taxon to be named (Huxley, 1865). Subsequently, Huene (1942) transferred “Ankistrodon indicus” to the genus “Chasmatosaurus” and referred to this species several vertebrae previously assigned to the dicynodont Dicynodon orientalis by Huxley (1865). Charig & Reig (1970) considered the holotype of “Ankistrodon indicus” indeterminate and, as a result, the genus and species should be regarded as nomina dubia, as previously proposed by Gower (1994). I agree with the conclusion of Charig & Reig (1970) after a first-hand study of the holotype, but the species was included in the taxonomic sampling of the present phylogenetic analysis in order to test its supposed proterosuchid affinities.

Tasmaniosaurus triassicus Camp & Banks, 1978

Age. Induan–early Olenekian, Early Triassic (Ezcurra, 2014).

Locality. Crisp and Gunn’s Quarry at the head of Arthur Street (42°52′50.0″S 147°18′10.6″E±100 metres), Hobart, Tasmania, Australia (Thulborn, 1986; Ezcurra, 2014).

Stratigraphic horizon. Upper levels of the Poets Road Siltstone Member, Knocklofty Formation, Upper Parmeener Supergroup, Tasmania Basin (Thulborn, 1986; Ezcurra, 2014).

Holotype. UTGD 54655: partial skeleton, mostly disarticulated, composed of right premaxilla; left maxilla; probable right maxilla; right lacrimal; both frontals, postfrontals and parietals; interparietal; ?supraoccipital; right pterygoid; ?epipterygoid; both dentaries; left splenial; one cervico-dorsal and one anterior or middle dorsal vertebra; fourteen to sixteen caudal vertebrae; several ribs, gastralia and haemal arches; interclavicle; ?femur; both tibiae; and multiple metatarsals and pedal phalanges (Ezcurra, 2014).

Supposed referred material. Some bone fragments from other localities in Tasmania were previously referred to Tasmaniosaurus triassicus (Warren, 1972; Forsyth et al., 1974) but could not be located in the collection of the UTGD in August of 2012. Thulborn (1986) considered these bones indeterminate.

Diagnosis. Tasmaniosaurus triassicus is a small-sized basal archosauromorph distinguished from other members of the clade by the following unique combination of character-states: premaxilla with posterodorsally oriented posterior process and ankylothecodont tooth implantation; maxilla with anteroposteriorly short anterior process; frontal with almost straight lateral margin; pterygoid with medial row of palatal teeth; dorsal vertebrae with paradiapophyseal and prezygodiapophyseal laminae and without distinct distal expansion of the neural spine; probable absence of osteoderms; and interclavicle with a diamond-shaped anterior end and a gracile and slightly transversely expanded posterior process (Ezcurra, 2014: 5).

Remarks. Camp & Banks (1978) named the proterosuchid species Tasmaniosaurus triassicus, which was redescribed by Thulborn (1986). This species was barely cited in the scientific literature during the subsequent 20 years (Ezcurra, Butler & Gower, 2013), until it was redescribed and figured in detail by Ezcurra (2014). Tasmaniosaurus triassicus has not been included in any quantitative phylogenetic analysis.

“Exilisuchus tubercularis” Ochev, 1979nomen dubium

Age. Early Olenekian, Early Triassic (Gower & Sennikov, 2000).

Locality. Kzyl-Sai III 2 locality, Akbulak district, Orenburg Province, Russia (Ochev, 1979; Gower & Sennikov, 2000).

Stratigraphic horizon. Sludkian Gorizont, Vetlugian Supergorizont (Ochev, 1979; Gower & Sennikov, 2000).

Holotype. PIN 4171/25: partial left ilium.

Remarks. “Exilisuchus tubercularis” was named by Ochev (1979) and interpreted as a possible proterosuchian. Sennikov (1995b) concluded that the systematic position of the species is very problematic, but it might represent a proterosuchid archosauriform. Gower & Sennikov (2000) and Ezcurra, Butler & Gower (2013) listed “Exilisuchus tubercularis” as a possible proterosuchid species. The isolated ilium lacks part of the preacetabular process, dorsal margin of the iliac blade and most of the postacetabular process (PIN 4171/25) (Fig. 9). The base of the preacetabular process possesses a tuberosity that does not reach the supraacetabular crest, resembling the condition in tanystropheids (e.g., Macrocnemus bassanii: Rieppel, 1989a; Tanystropheus longobardicus: Nosotti, 2007) and some dinosauriforms (e.g., Silesaurus opolensis: Dzik, 2003; Saturnalia tupiniquim: Langer, 2003). By contrast, this tuberosity is absent in early archosauriforms (e.g., Proterosuchus alexanderi: NMQR 1484; “Chasmatosaurus” yuani: IVPP V2719; Erythrosuchus africanus: NHMUK PV R3592). An anteroposteriorly concave depression is placed immediately posterior to the tuberosity of the preacetabular process. The supraacetabular crest is poorly laterally developed. The medial surface of the ilium possesses two distinct facets for articulation with the sacral ribs. The ventral margin of the acetabular wall is damaged, but ventrally convex as preserved.

Figure 9 “Exilisuchus tubercularis.”

Holotype partial left ilium (PIN 4171/25) in (A) lateral; (B) medial; and (C) dorsal views. Numbers indicate character-states scored in the data matrix and the arrows indicate anterior direction. Abbreviations: acw, acetabular wall; f.S1–2, facet for articulation with the first and second sacral ribs, respectively; isp, ischial peduncle; prap, preacetabular process; poap, postacetabular process; pup, pubic peduncle. Scale bar equals 2 mm.

The morphology of the holotype of “Exilisuchus tubercularis” differs from that present in proterosuchids and other archosauriforms but closely resembles that of tanystropheid archosauromorphs. Indeed, PIN 4171/25 cannot be distinguished from the ilia of Tanystropheus longobardicus and Macrocnemus bassanii and, as a result, “Exilisuchus tubercularis” is considered a nomen dubium, as previously proposed by Gower (1994). This species is included within the taxonomic sample of the current analysis in order to test quantitatively the phylogenetic position of the specimen.

“Blomosuchus georgii” (Sennikov, 1992) nomen dubium

Age. Induan, earliest Triassic (Sennikov, 1992).

Locality. Spasskoe I locality, Vetluga River, Nizhnii Novgorod Province, Russia (Sennikov, 1992).

Stratigraphic horizon. Vokhmian Gorizont (Sennikov, 1992).

Holotype. PIN 1025/348: partial parabasisphenoid.

Putative referred material. PIN 1025/14: partial parabasisphenoid.

Remarks. Two putative proterosuchid species have been named from the same locality (Spasskoe I) of the Early Triassic Vokhmian Gorizont of Russia: “Blomosuchus gregorii” and Vonhuenia friedrichi (Sennikov, 1992). Sennikov (1992) named “Blomosuchus georgii” on the basis of an isolated partial parabasisphenoid that lacks most of the cultriform process and part of the intertuberal plate and the dorsolateral surface of the bone (PIN 1025/348) (Fig. 10A). Subsequently, Sennikov (1995b) referred to “Blomosuchus georgii” some isolated postcranial bones from the same locality. The holotype of Vonhuenia friedrichi is an isolated posterior cervical vertebra (PIN 1025/11) (Figs. 10D–10F). An isolated parabasisphenoid, some vertebrae and other fragmentary postcranial bones collected in the type locality were referred to Vonhuenia friedrichi (Sennikov, 1992; Sennikov, 1995b) (Figs. 10B and 10G–10I).

Figure 10 “Blomosuchus georgii” and Vonhuenia friedrichi.

(A) holotype (PIN 1025/348) and (B) referred (PIN 1025/14) partial parabasisphenoids of “Blomosuchus georgii”; (C) parabasisphenoid of Proterosuchus alexanderi (NMQR 1484); and (D–F) holotype (PIN 1025/11) and (G–I) referred (PIN 1025/419, mirrored) cervico-dorsal vertebrae of Vonhuenia friedrichi in (A–C, F, I) ventral; (D) left lateral; (E, H) anterior; and right lateral (G) views. Numbers indicate character-states scored in the data matrix and the arrows indicate anterior direction. Abbreviations: af, accessory facet; btpbs, basal tubera of the parabasisphenoid; clp, clinoid process; cup, cultriform process; di, diapophysis; fo, foramina; pa, parapophysis; prz, prezygapophysis; vk, ventral keel. Scale bars equal 2 mm in (A, B, D–I) and 4 mm in (C).

The isolated partial parabasisphenoid (PIN 1025/14) referred to Vonhuenia friedrichi was distinguished from the holotype of “Blomosuchus gregorii” (PIN 1025/348) because of the presence of a proportionally longer parabasisphenoid body and differences in the shape of the basipterygoid processes (A Sennikov, pers. comm., 2013). However, the breakage of the median region of the intertuberal plate gives the artificial appearance that the body of the parabasisphenoid is anteroposteriorly shorter along the median line in the holotype of “Blomosuchus georgii” than it is in PIN 1025/14 (Figs. 10A and 10B). The right basipterygoid process of PIN 1025/14 is completely missing and only the base of the left process is preserved, with the latter having a strongly weathered surface. The position of the base of the basipterygoid process is identical in the holotype of “Blomosuchus georgii” and PIN 1025/14, and the orientation and shape of the distal articular surface of the basipterygoid process cannot be determined in PIN 1025/14. Accordingly, no substantial difference is recognised here between the holotype of “Blomosuchus georgii” and PIN 1025/14. As a result, both specimens probably belong to the same species, and a redescription and comparisons of these specimens with other early archosauriforms are provided as follows.

The parabasisphenoid of “Blomosuchus georgii” is horizontal in lateral view, constrasting with the posterodorsally-to-anteroventrally oriented braincases of Sarmatosuchus and erythrosuchids (Gower & Sennikov, 1996; Gower & Sennikov, 1997). The ventral surface of the basal tubera is strongly transversely convex and its lateral surface possesses a semilunar depression. This depression is slightly more anteriorly placed than that of Sarmatosuchus otschevi (Gower & Sennikov, 1997), but resembles the position present in Proterosuchus fergusi (NMQR 880) and Fugusuchus hejiapensis (Gower & Sennikov, 1996). The dorsal surface of the basal tubera is strongly transversely concave, and should have received the basioccipital contribution of the basal tubera and probably formed the floor of the unossified passage of the pseudolagenar recess. The intertuberal plate is developed as a posteriorly projected and curved shelf, which is completely preserved in the referred specimen (PIN 1025/14). The ventral surface of the parabasisphenoid possesses a shallow depression immediately anterior to the intertuberal plate. The foramina for the exit of the internal carotid arteries are placed posteromedially to the base of the basipterygoid processes, on the ventral surface of the braincase. These foramina are oval, with an anteromedially-to-posterolaterally oriented main axis. Well developed parasphenoid crests on the ventral surface of the parabasisphenoid originate posterior to the foramina for the passage of the internal carotids, curve anteriorly and run parallel to each other between the bases of the basipterygoid process and onto the base of the cultriform process. The morphology of the parasphenoid crests of “Blomosuchus georgii” is identical to that present in Prolacerta broomi (BP/1/2675), Proterosuchus alexanderi (NMQR 1484), Proterosuchus goweri (NMQR 880) and Fugusuchus hejiapensis (Gower & Sennikov, 1996). The parasphenoid crest is thicker on the cultriform process than more posteriorly. The basipterygoid processes are posteroventrally oriented in lateral view, as occurs in Proterosuchus alexanderi (NMQR 1484), Proterosuchus goweri (NMQR 880) and Fugusuchus hejiapensis (Gower & Sennikov, 1996). The groove for the palatine ramus of the facial nerve is very deep on the lateral surface of the bone and anteriorly defined by the clinoid process.

These comparisons between the holotype and probably referred parabasisphenoid of “Blomosuchus georgii” and other basal archosauriforms indicate that this species cannot be distinguished from the parabasisphenoids of other closely related species, such as Proterosuchus fergusi (BP/1/3993), Proterosuchus alexanderi (NMQR 1484), Proterosuchus goweri (NMQR 880) and Fugusuchus hejiapanensis (Gower & Sennikov, 1996). Therefore,“Blomosuchus” and “Blomosuchus georgii” are considered here as nomina dubia, as previously proposed by Gower (1994). Nevertheless, this species is included in the taxonomic sampling of the analysis to test quantitatively the phylogenetic affinities of the species for the first time and the presence of proterosuchids in the Lower Triassic beds of Russia. The lack of anatomical overlap between the postcranial bones previously referred to “Blomosuchus georgii” and the holotype, combined with the presence of another nominal species of putative proterosuchid (Vonhuenia friedrichi) at the same locality and horizon, suggests that caution is warranted in the taxonomic assignment of these specimens. As a result, the hypodigm of “Blomosuchus georgii” is restricted to and scored here based on the holotype (PIN 1025/348) and the parabasisphenoid PIN 1025/14 (previously referred to Vonhuenia friedrichi).

Vonhuenia friedrichi Sennikov, 1992

Age. Induan, earliest Triassic (Sennikov, 1992).

Locality. Spasskoe I locality, Vetluga River, Nizhnii Novgorod Province, Russia (Sennikov, 1992).

Stratigraphic horizon. Vokhmian Gorizont (Sennikov, 1992).

Holotype. PIN 1025/11: posterior cervical vertebra.

Referred material. PIN 1025/419: anterior portion of a posterior cervical vertebra.

Emended diagnosis. Vonhuenia friedrichi is a small-sized basal archosauromorph differentiated from the cervico-dorsal vertebrae of other members of the clade by the following unique combination of character-states: median keel on the ventral surface of the centrum; prezygodiapophyseal and postzygodiapophyseal laminae on the neural arch; accessory rib facet; and absence of mammillary processes on the neural spine.

Remarks. Sennikov (1992) erected the new genus and species Vonhuenia friedrichi on the basis of an isolated posterior cervical vertebra with three rib facets (Figs. 10D–10F) and referred to the species several fragmentary postcranial, isolated bones. Sennikov (1995b) referred additional isolated specimens from the type locality to this species. As was the case for “Blomosuchus georgii”, there is no direct overlap between the isolated posterior cervical vertebra that represents the holotype of Vonhuenia friedrichi (PIN 1025/11) and the specimens referred to this taxon, with the probable exception of a single vertebra that also possesses a third rib articular facet (PIN 1025/419) (Figs. 10G–10I). This condition is not present among the other referred vertebral specimens, which have only two articular facets for the ribs. This suggests that the other referred vertebrae are from different positions in the vertebral column than the holotype and PIN 1025/419. The latter specimen shares with the holotype (PIN 1025/11) the presence of prezygodiapophyseal and postzygodiapophyseal laminae, as well as a median longitudinal keel on the ventral surface of the centrum. This combination of features is not present in the cervico-dorsal vertebrae of other basal archosauromorphs and thus is diagnostic for Vonhuenia friedrichi and, as a result, the genus and species seems to be valid (contra Gower, 1994). The other previously referred vertebrae do not possess this combination of features, although it may be because they are from a different region of the vertebral column than PIN 1025/11 and PIN 1025/419. As a result, the hypodigm of Vonhuenia friedrichi is restricted here to the holotype (PIN 1025/11) and a referred cervico-dorsal vertebra (PIN 1025/419).

Chasmatosuchus rossicus Huene, 1940 (=“Tsylmosuchus samarensis” Sennikov, 1990)

Age. Early Olenekian, Early Triassic (Gower & Sennikov, 2000).

Locality. Vakhnevo locality, southern Cis-Urals, Vologda Oblast, Russia (Huene, 1940; Gower & Sennikov, 2000).

Stratigraphic horizon. Rybinskian Gorizont (Huene, 1940; Gower & Sennikov, 2000).

Lectotype. PIN 2252/381: two articulated posterior cervical vertebrae (ca. Cvs. 8–9) and their two respective intercentra.

Referred material. PIN 3200/217, 2424/6 (holotype of “Tsylmosuchus samarensis”): very anterior cervical vertebra (ca. Cv. 3); PIN 3200/472: anterior or middle cervical vertebra lacking most of the neural arch; PIN 3200/212: posterior dorsal vertebra; PIN 2243/167, 2252/384, 386: anterior caudal vertebrae lacking the transverse processes and distal portion of the neural spine.

Emended diagnosis. Small archosauromorph that differs from other diapsids in the following unique combination of features: vertebrae with a deep fossa present laterally to the base of the neural spine; anterior and middle cervical vertebrae with a shelf-like laterally flaring, thick tuberosity projected posteriorly from the base of the diapophysis along the lateral surface of the centrum, and an anterodorsally-to-posteroventrally oriented, thin lamina that delimits the anterolateral border of the fossa placed laterally to the base of the neural spine; and posterior cervical and dorsal vertebrae lacking an anteroventrally-to-posterodorsally oriented bulbous tuberosity placed on the centrodiapophyseal fossa (tuberosity present in Chasmatosuchus magnus).

Figure 11 Chasmatosuchus rossicus.

(A) holotype posterior cervical vertebrae (PIN 2252/381) and (B, C: PIN 3200/217; E, F: PIN 2424/6, holotype of Tsylmosuchus samariensis, mirrored) referred anterior cervical vertebrae; and Chasmatosuchus magnus (D, G) referred anterior cervical vertebra (PIN 3361/13, holotype of “Gamosaurus lozovskii,” mirrored) in left lateral (A, B), ventral (C, F, G), and right lateral (D, E) views. Numbers indicate character-states scored in the data matrix and the arrows indicate anterior direction. Abbreviation: dr, diagonal ridge. Scale bars equal 5 mm.

Remarks. Chasmatosuchus rossicus was named by Huene (1940) based on two articulated posterior cervical vertebrae (PIN 2252/381) (Fig. 11A) and a referred proximal end of a left tibia (PIN 2355/25). In the same contribution, Huene (1940) erected a second species for the genus, “Chasmatosuchus parvus,” based on an isolated probable middle cervical centrum (PIN 2252/382). However, this second species was synonymized with Chasmatosuchus rossicus by Tatarinov (1961) and referred to Microcnemus efremovi by Sennikov (1995b). The referral of the isolated proximal end of tibia to Chasmatosuchus rossicus is problematic because of the absence of overlapping features with the type specimen. As a result, this tibia is not included here within the hypodigm of Chasmatosuchus rossicus. Nevertheless, some cervical, dorsal and caudal vertebrae can be referred to Chasmatosuchus rossicus because of the presence of a unique combination of features not present in other archosauriforms (contra Gower, 1994), including a very deep, anteroposteriorly elongated fossa placed immediately lateral to the base of the neural spine. In addition, the cervical vertebrae of Chasmatosuchus rossicus possess an anterodorsally-to-posteroventrally oriented, thin ridge that delimits the anterolateral border of the fossa placed laterally to the base of the neural spine and extends onto the lateral surface of the base of the prezygapophysis. The presence of this lamina is only shared with “Tsylmosuchus samarensis,” named on the basis of an anterior-middle cervical vertebra (Figs. 11E and 11F) from the same stratigraphic level as Chasmatosuchus rossicus (“Tsylmosuchus samarensis” was considered a nomen dubium by Gower (1994)), and cervical and anterior dorsal proteroschid-like vertebrae collected in the Lower Triassic Panchet Formation of India (e.g., GSI 2111, 2116). However, the anterior and middle cervical vertebrae of Chasmatosuchus rossicus and “Tsylmosuchus samarensis” differ from the Panchet proterosuchid vertebrae in the presence of a shelf-like laterally flaring, thick tuberosity projected posteriorly from the base of the diapophysis along the lateral surface of the centrum, a condition that is only shared with Chasmatosuchus magnus and “Gamosaurus lozovskii” (Figs. 11B–11G). Therefore, the unique combination of characters present in Chasmatosuchus rossicus and “Tsylmosuchus samarensis” suggests that these species are synonymous. The presence of the shelf-like tuberosity is also present in the type anterior cervical and single known specimen of the type species of the genus Tsylmosuchus, Tsylmosuchus jakovlevi (PIN 4332/1), and strongly suggests that it is a proterosuchian-grade archosauriform rather than a suchian archosaur, contrasting with its original assignment (Sennikov, 1990). In addition, the third species of the genus, Tsylmosuchus donensis, is distinguished only tentatively from the other species because of their general anatomical similarities (Sennikov, 1990; Gower & Sennikov, 2000) and may also represent a proterosuchian-grade archsauriform. Tsylmosuchus jakovlevi and Tsylmosuchus donensis cannot be clearly distinguished from Chasmatosuchus rossicus and Chasmatosuchus magnus beyond some subtle features and probably are nomina dubia, as was previously proposed by Gower (1994).

Chasmatosuchus magnus Ochev, 1979 (=“Jaikosuchus” magnus (Ochev, 1979), =“Gamosaurus lozovskii” Ochev, 1979)

Age. Late Olenekian, late Early Triassic (Ochev, 1979; Gower & Sennikov, 2000).

Locality. Rassypnaya locality, Orenburg region, Russia (Ochev, 1979; Gower & Sennikov, 2000).

Stratigraphic horizon. Upper Yarenskian Gorizont (Ochev, 1979; Gower & Sennikov, 2000).

Holotype. PIN 951/65: anterior cervical vertebra.

Referred material. PIN 3361/13 (holotype of “Gamosaurus lozovskii”): anterior or middle cervical vertebra; PIN 3361/94, 183: two anterior or middle cervical vertebrae, respectively, lacking most of the neural arch; PIN 3361/14: posterior cervical vertebra; PIN 3361/213, 214: anterior dorsal vertebrae lacking most of the neural arch.

Emended diagnosis. Small archosauromorph that differs from other diapsids in the following combination of features: vertebrae with a deep fossa present laterally to the base of the neural spine; anterior and middle cervical vertebrae with a shelf-like laterally flaring, thick tuberosity projected posteriorly from the base of the diapophysis along the lateral surface of the centrum; and posterior cervical and dorsal vertebrae with an anteroventrally-to-posterodorsally oriented bulbous tuberosity placed on the centrodiapophyseal fossa.

Remarks. Two putative proterosuchids have been described from the upper Olenekian Yarenskian Gorizont of Russia: “Gamosaurus lozovskii” and Chasmatosuchus magnus (Ochev, 1979). The holotype of “Gamosaurus lozovskii” is an isolated partial anterior cervical vertebra (PIN 3361/13) (Ochev, 1979) (Figs. 11D and 11G), and Sennikov (1995b) referred to this species a middle (PIN 3361/213) and a posterior (PIN 3361/214) cervical vertebrae. The holotype of Chasmatosuchus magnus is an isolated anterior cervical vertebra (PIN 951/65), and Ochev (1979) referred to this species some vertebrae from the type horizon and the underlying Ustmylian Gorizont, as well as a fibula from the type locality and horizon. Subsequently, Sennikov (1990) erected the new genus “Jaikosuchus” for the species, resulting in the new combination “Jaikosuchus” magnus. He also restricted the referred material of this species to a single neural arch. “Gamosaurus lozovskii” is considered here a subjective junior synonym of Chasmatosuchus (=“Jaikosuchus”) magnus because the anterior cervical vertebrae of the former species are identical in morphology to the holotype of Chasmatosuchus magnus and they share the presence of a strongly developed, shelf-like tuberosity on the lateral surface of the centrum that extends posteriorly from the base of the diapophysis. This feature is not present in other basal archosauromorphs with the exception of Chasmatosuchus rossicus. The distinction between Chasmatosuchus magnus and Chasmatosuchus rossicus is problematic because their type specimens belong to different regions of the cervical series, but at least one feature (i.e., posterior cervical and dorsal vertebrae with an anteroventrally-to-posterodorsally oriented bulbous tuberosity placed on the centrodiapophyseal fossa) present in the lectotype of the latter species and absent in the former species suggests the presence of two different taxa (contra Gower, 1994). As a result, though the distinction between the species is relatively weak, it is followed here and should be tested in the future when further specimens become available. In order to test this hypothesis of synonymy both species (i.e., “Gamosaurus lozovskii” and Chasmatosuchus magnus) were scored as independent terminals in a first phylogenetic analysis and then merged together as a combined Chasmatosuchus magnus in a second phylogenetic analysis (see below).

“Chasmatosuchus” vjushkoviOchev, 1961

Age. Late Olenekian, late Early Triassic (Ochev, 1961; Gower & Sennikov, 2000).

Stratigraphic horizon. Yarenskian Gorizont, Russia (Ochev, 1961).

Holotype. PIN 2394/4: left premaxilla that lacks most of the prenarial process.

Emended diagnosis. Small archosauromorph that differs from other diapsids in the following unique combination of features: anteroposteriorly narrow base of the prenarial process of the premaxilla; large, subcircular foramina on the anterior surface of the premaxillary body; dorsal flange along the base of the postnarial process of the premaxilla that gradually inceases in height posteriorly; ankylothecodont tooth implantation; more than five tooth positions in the premaxilla; and serrations on both sides of the premaxillary tooth crowns.

Figure 12 “Chasmatosuchus” vjushkovi.

Holotype partial left premaxilla (PIN 2394/4) in (A) lateral and (B) medial views. Numbers indicate character-states scored in the data matrix and the arrows indicate anterior direction. Abbreviations: pap, palatal process; pnp, postnarial process; prp, prenarial process; rap, reabsorption pit. Scale bar equals 5 mm.

Remarks. Ochev (1961) erected the new species “Chasmatosuchus” vjushkovi based on an isolated premaxilla (Fig. 12), but this author stated that the assignment of this new species to the genus Chasmatosuchus was very tentative. The overall morphology of the holotype is very similar to that of other archosauriforms with a strongly downturned premaxilla, such as Archosaurus rossicus, Proterosuchus spp., and Sarmatosuchus otschevi (Ezcurra, Butler & Gower, 2013). Indeed, “Chasmatosuchus” vjushkovi shares with these species the presence of more than five premaxillary teeth with ankylotheocodont implantation, and a palatal process distinctly divergent from the main axis of the postnarial process, which indicates the presence of a strongly downturned premaxilla (PIN 2394/4). The postnarial process possesses a dorsal flange that increases gradually in height posteriorly, resembling the condition in Archosaurus rossicus (PIN 1100/55), Proterosuchus fergusi (BP/1/3993, RC 846), Proterosuchus goweri (NMQR 880), and “Chasmatosaurus” yuani (IVPP V4067, 36315). By contrast, the dorsal flange on the postnarial process of Sarmatosuchus otschevi increases abruptly in height posteriorly at its base and forms a distinct inflexion with the rest of the dorsal margin of the process in lateral view (PIN 2865/68-9). The prenarial process of “Chasmatosuchus” vjushkovi possesses an anteroposteriorly narrow base, as occurs in Sarmatosuchus otschevi (PIN 2865/68-9), but contrasting with the deeper base of the process present in Archosaurus rossicus (PIN 1100/55), Proterosuchus fergusi (BP/1/3993, RC 846), Proterosuchus goweri (NMQR 880), and “Chasmatosaurus” yuani (IVPP V4067, 36315). Therefore, “Chasmatosuchus” otschevi possesses a combination of features absent in other archosauriforms with a downturned premaxilla and seems to be a valid species based on the currently available evidence (contra Gower, 1994).

Proterosuchid from Long Reef

Age. Middle Early Triassic (Kear, 2009).

Locality. Long Reef, noth of Sydney, New South Wales, Australia (Kear, 2009).

Stratigraphic horizon. Bulgo Sandstone, Narrabeen Group, Sydney Basin (Kear, 2009).

Material. SAM P41754: two associated anterior dorsal vertebrae, one of them missing most of the neural arch.

Remarks. Kear (2009) reported the discovery of two associated archosauriform anterior dorsal vertebrae from the Early Triassic Sydney Basin, which represented the first occurrence of a diapsid in this stratigraphic unit. These vertebrae were interpreted as belonging to a proterosuchid because of their anteroposteriorly elongate centra, dorsal neural spines with height greater than length, the possible presence of intercentra, double-headed rib articulations and well-developed distal tables on the neural spines. This specimen was included in the present phylogenetic analysis to test quantitatively its affinities for the first time.

Koilamasuchus gonzalezdiazi Ezcurra, Lecuona & Martinelli, 2010

Age. Ladinian–Carnian, Middle–Late Triassic (Ottone et al., 2014). This age is based on a single radioisotopic date, but correlations based on vertebrate biostratigraphy suggest an Early Triassic age (Bonaparte, 1981; Martinelli, De la Fuente & Abdala, 2009; Ezcurra, Lecuona & Martinelli, 2010).

Locality. Agua de los Burros locality, 35 km south of the city of San Rafael, Mendoza Province, Argentina (Bonaparte, 1981; Ezcurra, Lecuona & Martinelli, 2010).

Stratigraphic horizon. Quebrada de Los Fósiles Formation, Puesto Viejo Group (Bonaparte, 1981; Ezcurra, Lecuona & Martinelli, 2010).

Holotype. MACN-Pv 18119: very well preserved natural external molds of three dorsal vertebrae, at least six osteoderms, a dorsal rib, a probable gastralium, a chevron, a humerus, a probable radius, an ilium, an incomplete ungual phalanx, two metapodial fragments, and some indeterminate bone fragments.

Diagnosis. Koilamasuchus gonzalezdiazi is a small diapsid (total length of ca. 50 cm) distinguished among archosauriforms by the following combination of features (autapomorphy indicated with as asterisk): dorsal vertebral centra with a deep, well-defined, and ovoid lateral fossa; dorsal neural spines moderately tall and sub-triangular in lateral view; dorsal rib with a laterally curved proximal end and a sharp medial inflection below it, deep longitudinal sulcus on the proximal two-thirds of the shaft, and holocephalous; humerus with strongly expanded proximal and distal ends; oblique tuberosity on the shaft of the humerus*; ilium with well-developed preacetabular process; and presence of osteoderms (Ezcurra, Lecuona & Martinelli, 2010: 1436).

Remarks. Bonaparte (1981) reported and briefly described the remains of a proterosuchid from the Early Triassic of western Argentina (but more recently reinterpreted as Middle–Late Triassic in age; Ottone et al., 2014). This specimen was mostly overlooked in the scientific literature until it was redescribed by Ezcurra, Lecuona & Martinelli (2010). These authors found a unique combination of features in MACN-Pv 18119 to support the erection of the new genus and species Koilamasuchus gonzalezdiazi. Ezcurra, Lecuona & Martinelli (2010) included this species in a quantitative phylogenetic analysis and recovered it as more crownward than the putative proterosuchids Proterosuchus spp., Sarmatosuchus otschevi and Fugusuchus hejiapanensis, and as the sister-taxon of erythrosuchids and more crownward archosauriforms.

Kalisuchus rewanensis Thulborn, 1979

Age. Induan, earliest Triassic (Warren & Hutchinson, 1990; Warren, Damiani & Yates, 2006).

Locality. The Crater locality, 11 km south of Rewan, central Queensland, Australia (Thulborn, 1979).

Stratigraphic horizon. Lower beds of the upper part of the Arcadia Formation, Rewan Group (Thulborn, 1979).

Holotype. QM F8998: partial left maxilla.

Emended diagnosis. Medium-sized archosauriform distinguished from other archosauromorphs by the following unique combination of features: absence of antorbital fossa on the ascending and horizontal processes of the maxilla; vertical ascending process of the maxilla with a very slightly concave anterior border of the antorbital fenestra; absence of a maxillo-nasal tuberosity; more than 14 maxillary tooth positions; and teeth with ankylothecodont tooth implantation.

Remarks. Kalisuchus rewanensis was erected by Thulborn (1979) on the basis of an isolated maxilla (QMF 8998) (Figs. 13A, 13C and 13D) and multiple referred cranial and postcranial isolated bones from different localities of the Arcadia Formation. The assignment of the referred specimens to Kalisuchus rewanensis is problematic because of the lack of overlapping features with the holotype. The referred specimens of Kalisuchus rewanensis that were available for study first-hand (most of the bones figured by Thulborn (1979) are not currently housed in the QMF collection and should be considered lost at present, M Ezcurra, pers. obs., 2012) possess a morphology consistent with that of a non-archosaurian archosauriform, but they do not possess a combination of apomorphies congruent with those expected in only one archosauriform subclade. Indeed, the holotype of Kalisuchus rewanensis seems to have an intermediate morphology between that of Proterosuchus spp. and erythrosuchids. As a result, the scorings for Kalisuchus rewanensis were based upon its holotype only, in order to avoid the artefacts that a chimaeric taxon composed of multiple non-closely-related taxa could potentially cause in character optimizations and ultimately in the tree topologies. A first-hand study of the holotype maxilla of Kalisuchus rewanensis (QM F8998) resulted in a substantial reinterpretation of the element and, therefore, it is redescribed as follows.

Figure 13 Kalisuchus rewanensis.

(A, C, D) Holotype partial left maxilla (QM F8998) and close up of an erupting posterior maxillary crown, and (B) formerly referred right pterygoid (QM F9521) in (A) lateral; (C) medial; and (B, D) ventral views. Numbers indicate character-states scored in the data matrix and the arrows indicate anterior direction. Abbreviations: ar, anterior ramus; asp, ascending process; f.ju, facet for articulation with the jugal; hp, horizontal process; lr, lateral ramus; ppr, palatal process; pr, posterior ramus. Scale bars equal 1 cm in (A–D), and 1 mm in the close up.

The holotype of Kalisuchus rewanensis was originally interpreted as a right maxilla by Thulborn (1979: 332, 333), with the presence of an unusual articulation with the premaxilla: “at the front of the maxilla a dorsal groove received the sub-narial ramus of the premaxilla in normal archosaur fashion, but lateral to this there was a more extensive secondary contact missing the posterior half.” However, the holotype is reidentified here as a left maxilla and the unusual secondary contact is reinterpreted as the palatal process (Fig. 13C: ppr), which closely resembles that present in other basal archosauriforms (e.g., Garjainia prima: PIN 951/55). Furthermore, a fragment of tooth bearing bone was located in the collection (collected from the same locality as the holotype but after the publication of Kalisuchus rewanensis; R. Thulborn field note housed with the specimen) that fits perfectly with the cleanly broken margin of the anterior half of the horizontal process. This distal half of the horizontal process preserves interdental plates on the same face of the bone as the shelf-like process, demostrating clearly that the latter structure is on the medial surface and that the element is a left maxilla (contra Thulborn, 1979).

The anterior tip of the maxilla is damaged, but the position of the palatal process indicates that only a very small portion of the anterior process is missing. As a result, the anterior process of the maxilla is proportionally very short in comparison with the length of the horizontal process, resembling the condition in an isolated archosauriform maxilla from the Early Triassic of South Africa (NMQR 3570), Fugusuchus hejiapanensis (Cheng, 1980) and erythrosuchids (e.g., Erythrosuchus africanus: BP/1/5207; Garjainia prima: PIN 2394/5-1). By contrast, the anterior process of the maxilla is proportionally longer in Proterosuchus fergusi (SAM-PK-11208; RC 846), Proterosuchus alexanderi (NMQR 1484), Proterosuchus goweri (NMQR 880), and “Chasmatosaurus” yuani (IVPP V4067, 36315). The anterodorsal margin of the anterior process of the maxilla in Kalisuchus rewanensis is broken and, as a result, the shape of the transition between the anterior and ascending processes cannot be determined. The anterior tip of the maxilla possesses a dorsoventrally concave and anteroventrally-to-posterodorsally oriented depression that represents part of the facet for reception of the postnarial process of the premaxilla. The maxilla curves slightly laterally in ventral or dorsal view, but to a lower degree than in Proterosuchus goweri (NMQR 880). The ascending process of the maxilla lacks its anterior margin and distal end. This process is very dorsoventrally tall and vertical, resulting in a broadly, very gently concave anterior border of the antorbital fenestra in lateral view, as occurs in the isolated archosauriform maxilla from South Africa (NMQR 3570), and contrasting with the more concave border present in other basal archosauriforms (e.g., Proterosuchus fergusi: SAM-PK-11208, RC 846; Fugusuchus hejiapanensis: Cheng, 1980; Erythrosuchus africanus: BP/1/5207). There is no antorbital fossa on the preserved portion of the ascending process (i.e., immediately anterior to the antorbital fenestra), neither on the horizontal process, as occurs in Proterosuchus spp. (e.g., Proterosuchus fergusi: SAM-PK-11208, RC 846; “Chasmatosaurus” yuani: IVPP V4067, 36315) and Fugusuchus hejiapanensis (Cheng, 1980). The base of the asending process is medially inset with respect to the anterior and horizontal processes and its lateral surface is anteroposteriorly convex, contrasting with the flatter lateral surface of the maxilla of proterosuchids (e.g., Proterosuchus fergusi: SAM-PK-11208, RC 846; “Chasmatosaurus” yuani: IVPP V4067, 36315). The ascending process of the maxilla of Kalisuchus rewanensis gradually narrows transversely, becoming laminar towards its distal end. Articular facets for the nasal and lacrimal are not preserved. The ventral border of the antorbital fenestra, on the horizontal process, is concave in lateral view, resembling the condition in other early archosauriforms with the exception of Tasmaniosaurus triassicus (Ezcurra, 2014).

The medial surface of the maxilla is dorsoventrally convex above the alveolar margin at the anterior end and on the base of the horizontal processes. The distal half of the horizontal process is flat. The palatal process is placed immediately above the alveolar margin of the bone and curved ventrally in medial view. As a result, the distal end of the process is distinctly anteroventrally oriented in a stronger degree than in erythrosuchids (e.g., Garjainia prima: PIN 951/55), which may indicate that the premaxilla of Kalisuchus rewanensis was strongly downturned. The dorsal surface of the palatal process possesses a series of longitudinal striations on its most anterior end that merge gradually with the rest of the surface of the bone posteriorly. The medial surface of the ascending process is anteroposteriorly convex immediately anterior to the border of the antorbital fenestra. This convex surface delimits posteriorly an extensive, shallowly concave fossa. A pair of large, medial foramina is placed below the ascending process and level with the mid-height of the horizontal process. The most anterior foramen opens anteroventrally and the most posterior one is more ventrally placed and opens mainly ventrally. The medial surface of the horizontal process possesses a faint facet that becomes dorsoventrally higher anteriorly and probably articulated with the palatine. The dorsal surface of the horizontal process possesses a laminar, laterally placed vertical lamina. This lamina delimits laterally a longitudinal groove on the posterior half of the process, which may have received the anterior process of the jugal. If this was the case, the maxilla and jugal possessed an extensive, diagonal suture, as occurs in Proterosuchus fergusi (SAM-PK-11208).

The alveolar margin of the maxilla is rather damaged and there are 14 preserved alveoli, but the most posterior alveoli are missing. The alveoli are oval in ventral view, with an anteroposterior main axis. Partial teeth are preserved in the fourth, sixth, tenth and twelfth alveoli, but the most complete crown is placed in the fourth tooth socket. The teeth are ankylosed to the bone and the crowns are labiolingually compressed, distally recurved and with distal denticles along most of the margin and mesial denticles restricted to the apical half, resembling the condition in other archosauriforms (Nesbitt, 2011). Interdental plates are preserved on the posterior half of the horizontal process, though they are not well preserved. The interdental plates seem to have been sub-triangular to pentagonal, and their medial surfaces are ornamented with multiple pits.

One of the referred specimens that also deserves comment is the bone interpreted by Thulborn (1979) as a left jugal (QM F9521) (Fig. 13B). Thulborn (1979) thought that this putative jugal possessed a very strange morphology, with a ventrolateral projection below the anterior border of the orbit and a suture with the postorbital inverted from the common diapsid condition (i.e., with the postorbital articulating posteriorly to the jugal). This bone is reinterpreted here as a right pterygoid, with a morphology completely congruent with that present in other archosauriforms, such as Erythrosuchus africanus (NHMUK PV R3592) and Sarmatosuchus otschevi (PIN 2865/68-2). QM F9521 preserves the base of the anterior ramus, most of the posterior ramus and a complete lateral process. The facet interpreted by Thulborn (1979) as the articulation with the postorbital is reinterpreted as the facet to receive the ectopterygoid, whereas the supposed unusual ventrolateral flange is reinterpreted as the projection that partially wraps the distal end of the basipterygoid process of the parabasisphenoid in the basal articulation. The ventral surface of the base of the anterior ramus was originally partially covered with matrix, but after some repreparation of the specimen it was possible to determine the absence of palatal teeth, resembling the condition present in erythrosuchids (Gower, 2003; Ezcurra, Butler & Gower, 2013).

Fugusuchus hejiapaensis Cheng, 1980

Age. Late Olenekian–early Anisian, Early–Middle Triassic (Cheng, 1980).

Locality. Fugu County, Shanxi Province, People’s Republic of China (Cheng, 1980).

Stratigraphic horizon. Heshanggou Formation (Cheng, 1980).

Holotype. GMB V 313: fairly complete skull, an intercentrum, ulna, radius, and partial manus (modified from Parrish, 1992).

Emended diagnosis. Medium-sized archosauriform distinguished from other archosauromorphs by the following unique combination of features: anterior process of the maxilla shorter than the antorbital fenestra; less than 20 tooth positions in the maxilla; absence of maxillo-nasal tuberosity; posterior process of jugal with a semi-circular ventral expansion; squamosal forming a broadly concave posterodorsal corner of the infratemporal fenestra; otoccipial with a deep, teardrop-shaped fossa laterally to the foramen magnum; and sub-horizontal parabasisphenoid in lateral view.

Remarks. Cheng (1980) erected Fugusuchus hejiapaensis and provided a description in Chinese of the specimen. This author assigned Fugusuchus hejiapaensis to the Proterosuchidae, but subsequently Parrish (1992) recovered this species as the most basal member of Erythrosuchidae in a cladistic analysis. Gower & Sennikov (1996) redescribed and figured in detail the braincase of this species and recovered it as the sister-taxon of Proterosuchus spp. based on a phylogenetic analysis restricted to braincase characters. The same result was also recovered by Gower & Sennikov (1997) using characters of the entire skeleton. Ezcurra, Lecuona & Martinelli (2010) found Fugusuchus hejiapaensis in a trichotomy composed of Sarmatosuchus otschevi and other archosauriforms to the exclusion of Proterosuchus spp., thus resulting in a paraphyletic Proterosuchidae. Gower (1994) stated that the premaxilla indentified by Cheng (1980) is poorly preserved, incomplete and even possibly misindentified. In addition, Gower (1994) did not include the putative quadratojugal in his revised reconstruction of the skull of Fugusuchus hejiapaensis. The quadratojugal is reinterpreted here as an ectopterygoid (GMB V 313, unpublished photographs).

Sarmatosuchus otschevi Sennikov, 1994

Age. Anisian, early Middle Triassic (Tverdokhlebov et al., 2003; Lucas, 2010).

Locality. Locality Berdyanka 11, near Mikhailovka village, Berdyanka River basin, Sol-Iletsk district, Orenburg region, southern Cis-Urals, Russia (Gower & Sennikov, 1997).

Stratigraphic horizon. Upper part of the Donguz Gorizont (Gower & Sennikov, 1997).

Holotype. PIN 2865/68: partial skeleton of a single individual, including right premaxilla, right squamosal, incomplete right frontal, incomplete right parietal, right and left quadrates and incomplete jugals, left pterygoid and palatine, braincase without laterosphenoid and cultriform process, right dentary, both probable splenials, right prearticular, isolated teeth, second to eleventh presacral vertebrae, two isolated dorsal centra, one caudal vertebra, ribs, left and right scapulocoracoid, left ulna, and unidentified bone fragments (modified from Gower & Sennikov, 1997).

Diagnosis. Gower & Sennikov (1997: 62) diagnosed Sarmatosuchus otschevi as an archosauriform with estimated total body length of approximately two metres and distinguished from other archosauromorphs by the following unique combination of features: premaxilla downturned; jugal with slender anterior, and broad, semi-elliptical posterior processes; large foramen between quadrate and quadratojugal; teeth present on palatine and palatal ramus of pterygoid, but absent from posteroventral flange of pterygoid; braincase relatively short and high; intercentra present; centra of cervicals as tall as they are long; pectoral ribs three-headed; and scapula and coracoid relatively short and broad. In addition the following autapomorphy distinguishes Sarmatosuchus otschevi from other archosauromorphs: premaxilla with a dorsal flange on the postnarial process that increases abruptly in height posteriorly at its base and forms a distinct inflexion with the rest of the dorsal margin of the process in lateral view.

Remarks. Sennikov (1994) erected Sarmatosuchus otschevi based on a partial skeleton that represents the most complete specimen of a putative proterosuchid from Russia. This species was subsequently described in detail and included for the first time in a quantitative phylogenetic analysis by Gower & Sennikov (1997). These authors recovered Sarmatosuchus otschevi within a monophyletic Proterosuchidae, together with Proterosuchus spp. and Fugusuchus hejiapanensis. More recently, Ezcurra, Lecuona & Martinelli (2010) found Sarmatosuchus otschevi as the sister-taxon of erythrosuchids and more crownward archosauriforms.

Guchengosuchus shiguaiensis Peng, 1991

Age. Late Olenekian–early Anisian, late Early–early Middle Triassic (Peng, 1991; Fröbisch, 2009).

Locality. Gucheng, Fugu County, Shanxi Province, People’s Republic of China (Peng, 1991).

Stratigraphic horizon. Lower Ermaying Formation (Peng, 1991).

Holotype. IVPP V8808: left maxilla, partial skull roof, left pterygoid, partial braincase, posterior portion of the right hemimandible, two anterior–middle cervical vertebrae; one probable anterior dorsal lacking most of the centrum; a fragment of presacral vertebra; four cervical and dorsal ribs; partial right scapula, humerus, ulna, and radius, a metatarsal, and an ungual pahalanx.

Emended diagnosis. Medium-sized archosauriform distinguished from other archosauromorphs by the following unique combination of features (autapomorphies indicated with an asterisk): secondary antorbital fenestra formed by maxilla, nasal and probably premaxilla; maxilla with 14 tooth positions and ankylothecodont tooth implantation; maxilla without maxillo-nasal tuberosity and antorbital fossa; pterygoid without palatal teeth; anterior–middle cervical vertebrae with a strongly transversely convex and rugose distal expansion of the neural spine*; three-headed cervico-dorsal rib; and scapular blade with strongly concave posterior margin.

Remarks. Guchengosuchus shiguaiensis was erected by Peng (1991) based on a partial skeleton. This species has not been frequently mentioned in the scientific literature and has never been included in a quantitative phylogenetic analysis. Unfortunately, several of the bones originally figured and described by Peng (1991) could not be located in the collection of the IVPP and may be lost (scapula and limb bones), and other bones have been damaged (maxilla and pterygoid) since the original description. A detailed redescription of Guchengosuchus shiguaiensis is currently in preparation by the author and colleagues.

Cuyosuchus huenei Reig, 1961

Age. Late Carnian–early Norian, early Late Triassic (Spalletti, Fanning & Rapela, 2008; Ezcurra, Butler & Gower, 2013).

Locality. Cerro Bayo locality, Bajada de la Obligación, southwest of Mendoza city, Las Heras Department, Mendoza Province, Argentina (Rusconi, 1951).

Stratigraphic horizon. Cacheuta Formation, Cuyo Basin (Rusconi, 1951).

Holotype. MCNAM 2669: partial skeleton, including a left jugal, cervical, dorsal, sacral and caudal vertebrae, scapular and pelvic girdle bones and forelimb and hindlimb elements.

Diagnosis. Desojo, Arcucci & Marsicano (2002: 144) rediagnosed Cuyosuchus huenei as an archosauriform distinguished from other archosauromorphs by the following unique combination of features (autapomorphies marked with an asterisk): cervical and first dorsal vertebrae with anterior articular laminae that limit a cavity on each side of the neural arch; first to third caudal vertebrae the tallest of the vertebral column*; sub-rectangular scapular blade, well expanded anteroposteriorly, with the presence of a rounded coracoid; low iliac blade; subcircular, finely pitted, ventral osteoderms lacking articular facets*.

Remarks. Cuyosuchus huenei from the Late Triassic of western Argentina was originally assigned to the temnospondyl Chigutisaurus (Rusconi, 1951), but later reinterpreted as a proterosuchian thecodont and named by Reig (1961). Subsequently, Tatarinov (1961) and Hughes (1963) considered Cuyosuchus huenei an erythrosuchid and a junior synonym of Erythrosuchus africanus. However, Charig & Reig (1970: Fig. 6) interpreted Cuyosuchus huenei as a distinct erythrosuchid and, more recently, Desojo, Arcucci & Marsicano (2002) considered it as an archosauriform closer to crown archosaurs than to erythrosuchids. Thus, Cuyosuchus huenei would be among the youngest known non-archosaurian archosauriforms, but it has not yet been included in a quantitative phylogenetic analysis. A partial left jugal was identified among the indeterminate bones of the holotype during a recent first-hand restudy of the specimen. A detailed redescription of the species is currently in preparation by the author and colleagues.

Garjainia prima Ochev, 1958

Age. Late Olenekian, late Early Triassic (Ochev, 1958; Gower & Sennikov, 2000).

Locality. Kzyl-Sai (=Kzyl-Say) II 2 locality (locality 29 of Tverdokhlebov et al. (2003)), 0.5–1 km west of the village Andreevka, Akbulak district, Orenburg Province, Russia (Ochev, 1958; Gower & Sennikov, 2000).

Stratigraphic horizon. Petropavlovskaya Svita, Yarengian (=Yarenskian) Gorizont (Ochev, 1958; Gower & Sennikov, 2000).

Holotype. PIN 2394/5 (formerly SGU 104/3-43): partial skeleton of a single individual, including an almost complete skull (PIN 2394/5-1–5-7) and lower jaw (PIN 2394/5-8, 5-9), second to fifth cervical vertebrae (PIN 2394/5-10–5-13), two posterior cervical or anterior dorsal vertebrae (PIN 2394/5-16), at least four dorsal vertebrae (PIN 2394/5-14, 5-15, 5-17–5-19), both scapulae and coracoids (PIN 2394/5-32, 5-33), right clavicle (PIN 2394/5-35), interclavicle (PIN 2394/5-34), left fourth metatarsal (PIN 2394/5-36), several presacral ribs (PIN 2394/5-21–5-31), probable gastralia (PIN 2394/5-37), and some indeterminate fragments of bone (PIN 2394/5-37).

Referred material. Multiple specimens housed at the PIN, which previously composed the hypodigm of “Garjainia (=“Vjushkovia”) triplicostata” (Gower & Sennikov, 2000).

Emended diagnosis. Garjainia prima is a medium-sized erythrosuchid distinguished from other archosauromorphs by the following unique combination of character-states (autapomorphies indicated with an asterisk): premaxilla with a longitudinal groove on the lateral surface of the premaxillary body; nasal with an anteroposteriorly long descending process that forms an extensive longitudinal suture with the maxilla*; antorbital fossa absent on the horizontal process of the maxilla; antorbital fenestra trapezoidal and with an anteroventrally-to-posterodorsally orientated main axis*; prefrontal strongly flared laterally in dorsal view*; skull roof with a longitudinal fossa on its dorsal surface that harbours a longitudinal median prominence in its posterior half*; straight suture between postfrontal and postorbital; basioccipital with a median tuberosity on its ventral surface*; trigeminal cranial nerve (CN V) completely enclosed by prootic*; dentary with a posterodorsal process longer than the central posterior process; and interclavicle with a rhomboidal posterior ramus*.

Remarks. Ochev (1958) erected the new erythrosuchid genus and species Garjainia prima based on a fairly complete skull and associated partial postcranium from the late Early Triassic of Russia. This author described the general anatomy of the species in a series of papers (Ochev, 1958; Ochev, 1975; Ochev, 1981), but a detailed, comprehensive description is still missing. Huene (1960) named “Vjushkovia triplicostata”, a second erythrosuchid genus and species from the same gorizont as Garjainia prima. This new species was based on multiple and very well preserved cranial and postcranial specimens (Huene, 1960). Tatarinov (1961) proposed that both Garjainia and “Vjushkovia” were subjective junior synonyms of Erythrosuchus, and this hypothesis was followed by several later authors (e.g., Hughes, 1963; Ewer, 1965; Romer, 1966; Romer, 1972a; Cruickshank, 1972). However, this synonymy was rejected by Young (1964) and Charig & Reig (1970), and more recent authors (e.g., Parrish, 1992; Sennikov, 1995a; Sennikov, 1995b; Gower & Sennikov, 1996; Gower & Sennikov, 1997; Gower & Sennikov, 2000; Desojo, Arcucci & Marsicano, 2002; Ezcurra, Lecuona & Martinelli, 2010; Ezcurra, Butler & Gower, 2013; Ezcurra, 2014; Ezcurra & Butler, 2015a). Gower & Sennikov (2000) concluded that “Vjushkovia” is a subjective junior synonym of Garjainia, in agreement with previous comments by Kalandadze & Sennikov (1985), Ochev & Shishkin (1988), Sennikov (1995a) and Sennikov (1995b), and that “Garjainia triplicostata” is possibly also a subjective junior synonym of Garjainia prima. This taxonomic decision was recently followed by Gower et al. (2014). Indeed,“Garjainia triplicostata” differs from Garjainia prima only in the absence of palatal teeth on the pterygoid and palatine and the presence of completely thecodont tooth implantation. Both characters may be intraspecificlly variable and the hypothesis of synonymy is followed here because of the extremely similar morphology between the two nominal species. For example, the presence or absence of palatal teeth has been found to be intraspecifically variable in several lepidosauromorph species (Mahler & Kearney, 2006). Therefore, the scorings of Garjainia prima were based on the holotype of the species as well as the hypodigm of “Garjainia triplicostata.”

Garjainia madiba Gower et al., 2014

Age. Late Olenekian, late Early Triassic (Hancox, 2000).

Localities. Farm Driefontein 11, approximately 36 km NE of Senekal and 14 km N of Paul Roux, Thabo Mofutsanyane district municipality, Free State, South Africa (type locality); and several localities to the east and northeast of Senekal, in the same stratigraphic horizon as the holotype, Free State, South Africa (Gower et al., 2014).

Stratigraphic horizon. Burgersdorp Formation, Cynognathus AZ Subzone A, Beaufort Group, Karoo Supergroup, Karoo Basin (Gower et al., 2014).

Holotype. BP/1/5760: right postorbital, right postfrontal and partial frontal in articulation; ventral end of left postorbital; left jugal; right paroccipital process with partial supraoccipital and parts of right prootic in articulation; partial rib; at least four unidentified fragments, some of which likely represent partial skull elements.

Paratypes. Approximately 80 specimens from various localities listed by Gower et al. (2014) and housed at the BP and NMQR palaeontological collections.

Referred material. Sixteen specimens from various localities listed by Gower et al. (2014) and housed at the BP palaeontological collection.

Diagnosis. Gower et al. (2014: 6) distinguished Garjainia madiba from its only congener (Garjainia prima) in having the following character-states (autapomorphies marked with an asterisk): large bosses on the lateral surfaces of the jugal and postorbital*; dorsal end of the quadratojugal somewhat thickened; higher tooth counts for the premaxilla (six versus five) and maxilla (likely more than 14 versus 13 or 14); a longer postacetabular process of the dorsal blade of the ilium* (approximately as long as the acetabular portion of the ilium versus clearly shorter than the acetabular portion of the ilium).

Remarks. Hancox et al. (1995) and Hancox (2000) reported the presence of a Garjainia-like erythrosuchid in the upper Olenekian Cynognathus AZ, Subzone A, of South Africa. Gower et al. (2014) formally described these new erythrosuchid specimens and used them as the basis to erect the new species Garjainia madiba. These authors noted multiple similarities with the Russian erythrosuchid Garjainia prima and discussed qualitatively the phylogenetic relationships of the species. This species is included here in a quantitative phylogenetic anlaysis for the first time.

The holotype of Garjainia madiba (BP/1/5760) is based on cranial bones that probably belong to a single individual, and approximately 80 specimens were considered paratypes and 16 referred material (Gower et al., 2014). Most of the paratype and referred specimens of Garjainia madiba do not preserve overlapping features that can be compared with the holotype and, as a result, their assignment should be considered with caution. Nevertheless, all the bones referred to this species possess morphologies consistent with that of an animal very similar to Garjainia prima (Gower et al., 2014). As was the case for Kadimakara australiensis, the assignment of paratype and referred specimens to Garjainia madiba is taken here as a working hypothesis that will be tested in this phylogenetic analysis and by future discoveries of more complete, articulated specimens of the species.

Erythrosuchus africanus Broom, 1905

Age. Early Anisian, early Middle Triassic (Hancox, 2000).

Localities. Oorlogsfontein, Kraai River, a few miles east of Aliwal North, Eastern Cape Province, South Africa (type locality); and multiple localities from the type horizon of the species, South Africa (Broom, 1905; Gower, 2003).

Stratigraphic horizon. Burgersdorp Formation, Cynognathus AZ Subzone B, Beaufort Group, Karoo Supergroup, Karoo Basin (Broom, 1905; Gower, 2003).

Holotype. SAM-PK-905: partial postcranial skeleton.

Referred material. Multiple partial skeletons, cranial and postcranial bones housed in the AMNH, BP, GHG, MNHN, NHMUK PV, NMQR, SAM and UMCZ palaeontological collections. All referred specimens are listed by Gower (2003: Appendix I), and the following specimen is added here: MNHN 1869-12, right humerus.

Emended diagnosis. Erythrosuchus africanus is a large archosauriform distinguished from other archosauromorphs by the following unique combination of character-states: skull without a secondary antorbital fenestra; premaxilla with a peg on the posterior edge of the body; postorbital with incised groove on the posterior process; squamosal with a posterodorsally-to-anteroventrally oriented tuck on the lateral surface of the ventral process; quadratojugal very nearly or completely excluded from the infratemporal fenestra by a squamosal-jugal contact; foramen absent between quadrate and quadratojugal; parietal with posteromedial tubercle on the occipital surface of the posterolateral process; pterygoid without palatal teeth; basioccipital and parabasisphenoid excluded from the floor of the foramen magnum; ventral ramus of the opisthotic recessed within stapedial groove; stapedial groove with small bulge; medial wall of otic capsule not fully ossified; metotic foramen undivided; intercentra in presacral vertebral column; dorsal neural arches pierced by subdivided foramina; three-headed ribs in pectoral region; femur with internal trochanter positioned slightly ventral to the proximal end; astragalus spherical; calcaneum plate-like; fourth distal tarsal with ventral peg; first two distal tarsals and centrale absent; and metatarsal III longer than the others (modified from Gower, 2003: 11, 13).

Remarks. See historical background for the species in Gower (2003: 5, 7).

GHG 7433MI

Age. Late Olenekian–early Anisian, late Early–early Middle Triassic (Hancox, 2000).

Occurrence. Burgersdorp Formation, Cynognathus AZ (indeterminate subzone), Beaufort Group, Karoo Supergroup, Karoo Basin, South Africa. Unfortunately, the exact geographical and stratigraphical occurrence of the specimen is unknown.

Material. GHG 7433MI: fairly complete skeleton, including multiple disarticulated cranial bones and an articulated postcranium missing the distal half of the tail.

Remarks. Gower (2003) tentatively referred GHG 7433MI to Erythrosuchus africanus, but he did not justify this assignment. GHG 7433MI is a small erythrosuchid specimen, in comparison with the largest known specimens of Erythrosuchus africanus, with a total preserved length of around 60 cm and a complete total length estimated at around 1 metre. This specimen is the most complete, articulated erythrosuchid skeleton known so far. The bones of GHG 7433MI, mainly the cranial elements, are still covered with matrix and, as a result, several features cannot be determined or assessed confidently. The right maxilla seems to possess a palatal process placed immediately above the alveolar margin of the bone, resembling the condition of Garjainia prima (PIN 951/55) and Garjainia madiba (BP/1/5525), but contrasting with the more dorsally placed palatal process of Erythrosuchus africanus (BP/1/4680) and Shansisuchus shansisuchus (Young, 1964). However, the anterior end of the right maxilla is partially covered with matrix and this observation should be checked in the future after further preparation. The horizontal process of this maxilla increases abruptly in height towards its posteror end and results in an anteroventrally-to-posterodorsally slanting ventral border of the antorbital fenestra, as occurs in Garjainia prima (PIN 2394/5-1), but contrasting with the more horizontal ventral border of the opening of Erythrosuchus africanus (BP/1/5207). There is a large, subrectangular pineal fossa that extends on the dorsal surface of the frontals and parietals, resembling the condition in Erythrosuchus africanus (BP/1/5207), but contrasting with the oval fossa present in Garjainia prima (PIN 2394/5-1) and Shansisuchus shansisuchus (Young, 1964). Immediately anterior to the pineal fossa, there is a circular tuberosity on the median line of the frontals, which is not present in other archosauriforms of which I am aware. The parietal of GHG 7433MI possesses a transversely narrow supratemporal fossa, as occurs in Garjainia prima (PIN 2394/5-1) and Garjainia madiba (BP/1/5525), but contrasting with Erythrosuchus africanus and Shansisuchus shansisuchus, which lack this fossa (Young, 1964; Gower, 2003).

Based on the few comparisons listed above, it seems that GHG 7433MI belongs to a different taxon than Erythrosuchus africanus, but resembles both species of Garjainia in its morphology. A considerable amount of work is needed on this specimen to determine its taxonomic affinities and may shed light on the general body plan of erythrosuchids.

Shansisuchus shansisuchus Young, 1964

Age. Late Anisian, Middle Triassic (Rubidge, 2005; Ezcurra, Butler & Gower, 2013).

Localities. Hsishihwa 56173, Lotzeyue, Wuhsiang County, Shanxi Province, People’s Republic of China (type locality); several localities from the same horizon as the type locality, Wushiang, Ningwu, Yushe and Jingle counties of Shanxi Province, Xinxiang County of the Henan Province, and Jixian County of Heilongjiang Province, People’s Republic of China (Young, 1964; Wang et al., 2013).

Stratigraphic horizon. Upper member of the Ermaying Formation (Young, 1964; Wang et al., 2013).

Holotype. IVPP V2503: skull roof missing the anterior tip of the rostrum.

Paratypes. IVPP V2501, 2502, 2504–2511, 2512, 2513, 2540, 2593: partial skeletons (Young, 1964; Wang et al., 2013).

Referred material. SXMG V 00002: anterior third of an articulated skeleton, including a skull and a series of 14 vertebrae.

Diagnosis. Wang et al. (2013: 1187) diagnosed Shansisuchus shansisuchus as a large erythrosuchid differing from others in the following combination of character-states: six premaxillary teeth; a large subnarial fenestra anterior to the antorbital fenestra; tongue-in-groove articulations between the postnarial process of the premaxilla and nasal and between the premaxilla and maxilla; ascending process of maxilla tall, narrow, and posterodorsally extended; ventral process of the postorbital with a semi-circular projection into the orbit; broad ventral process of squamosal distally forked; and a large, deeply bow-shaped intercentrum tightly anchoring/capping the sharp ventral edges of two neighboring centra together in cervical and at least first eight dorsal vertebrae.

Remarks. Shansisuchus shansisuchus was named and originally described by Young (1964) based on multiple specimens from the Middle Triassic of China. The original description of the species was complemented by detailed descriptions of its braincase and ankle anatomy (Gower, 1996; Gower & Sennikov, 1996), and, more recently, a well-preserved partial, articulated skeleton was reported by Wang et al. (2013). Shansisuchus shansisuchus was included in several phylogenetic analyses focused on proterosuchian archosauriforms and was recovered in all of them as an erythrosuchid (Parrish, 1992; Gower & Sennikov, 1996; Gower & Sennikov, 1997; Ezcurra, Lecuona & Martinelli, 2010).

Shansisuchus kuyeheensis Cheng, 1980

Age. Late Anisian, Middle Triassic (Rubidge, 2005; Ezcurra, Butler & Gower, 2013).

Locality. Hejiachuan, Shenmu County, Shanxi Province, People’s Republic of China (Cheng, 1980).

Stratigraphic horizon. Upper member of the Ermaying Formation (Cheng, 1980).

Holotype. IGCAGS V 314: partial skeleton, including premaxilla, maxilla, dentary, most of the presacral vertebral series, scapula, coracoid and humerus.

Remarks. This species of Shansisuchus was erected and briefly described by Cheng (1980). Gower (1996) stated that there was no strong evidence to consider this species a valid taxon. Indeed, it was not possible to recognize any unique combination of features based on the description and figures of Cheng (1980) and this species might be a nomen dubium. Unfortunately, the holotype and only known specimen of Shansisuchus kuyeheensis was not studied at first-hand because could not be located in its collection and, as a result, its taxonomic validity will not be assessed here.

Chalishevia cothurnata Ochev, 1980

Age. Ladinian, late Middle Triassic (Shishkin et al., 2000).

Localities. Bukobay VII locality, Sol’-Iletsk distrinct, Orenburg Province, Russia (type and paratype localities); Koltaevo locality, Orenburg Province, Russia (paratype locality) (Ochev, 1980; Gower & Sennikov, 2000).

Stratigraphic horizon. Bukobay Gorizont (Ochev, 1980; Gower & Sennikov, 2000).

Holotype. PIN 4366/1 (wrongly cited as PIN 4356/1 by Gower & Sennikov, 2000): left maxilla and both nasals.

Paratypes. PIN 4366/2: partial right quadrate; PIN 4366/3, 8: teeth; PIN 2867/18: partial right nasal.

Referred material. PIN 2867/7: partial right nasal.

Emended diagnosis. Chalishevia cothurnata is a large erythrosuchid that differs from other archosauromorphs in the following combination of features (autapomorphy indicated with an asterisk): accessory antorbital fenestra and fossa*; maxillo-nasal tuberosity; and maxilla with an edentulous anterior tip, mainly vertical ascending process, and oblique, anteroventrally-to-posterodorsally oriented ventral border of the antorbital fossa on the horizontal process*.

Remarks. See comments in Gower & Sennikov (2000: 149).

Youngosuchus sinensis (Young, 1973b)

Age. Anisian–Ladinian, Middle Triassic (Lucas, 2010).

Occurrence. Kelamayi Formation, Xinjiang Autonomous Region, People’s Republic of China (Young, 1973b).

Holotype. IVPP V3239: complete skull and lower jaw, cervical vertebrae, and partial pectoral girdle and forelimbs.

Emended diagnosis. Youngosuchus sinensis is a medium-sized archosauriform that differs from other archosauromorphs in the following combination of features (autapomorphy indicated with an asterisk): skull without secondary antorbital fenestra and maxillo-nasal tuberosity; premaxilla with four tooth positions; maxilla contributes to the border of the external naris; horizontal process of the maxilla with a straight ventral border of the antorbital fenestra; maxilla with an antorbital fossa on the ascending and horizontal processes, but interrupted on the anterior border of the antorbital fenestra; nasal with a dorsally elevated anterior end above the skull roof, giving the skull a “Roman nose” appearance; lateral surface of the jugal with a plate-like prominence, separated anteroventrally from the rest of the bone by a distinct shelf and placed immediately below the ascending process*; neural spines of the cervical vertebrae without spine table; three-headed cervico-dorsal ribs; scapula with a prominent tuber on the posterior edge, just dorsal to the glenoid fossa; and coracoid with a rather well-developed and laterally expanded posterior process.

Remarks. Young (1973b) described a partial skeleton of a putative erythrosuchid from the Middle Triassic of China and used it as the basis for the new species “Vjushkovia” sinensis. Subsequently, Kalandadze & Sennikov (1985) reinterpreted this species as a rauisuchid archosaur and transferred it to the new genus Youngosuchus. Nevertheless, Parrish (1992) agreed with the original phylogenetic conclusion of Young (1973b) and returned to the generic assignment of “Vjushkovia” sinensis. Indeed, Parrish (1992) found “Vjushkovia triplicostata” and “Vjushkovia” sinensis as more closely related to each other than to other erythrosuchids, and Garjainia prima as the most basal erythrosuchid. Gower & Sennikov (2000) concluded that there is no sound morphological basis for a generic distinction between Garjainia and “Vjushkovia”, and that Garjainia prima and “Vjushkovia triplicostata” probably represent a single species. Therefore, these conclusions clearly contradicted the quantitative results recovered by Parrish (1992). After a first-hand study of all of these specimens, the interpretations of Gower & Sennikov (2000) are followed here, and “Vjushkovia” is considered a junior synonym of Garjainia. As a result, the taxonomic assignment of IVPP V3239 as Youngosuchus sinensis is used here.

“Dongusia colorata” Huene, 1940nomen dubium

Age. Anisian, early Middle Triassic (Tverdokhlebov et al., 2003; Lucas, 2010).

Locality. Donguz I locality, Sol’-Iletsk district, Orenburg Province, Russia (Huene, 1940; Gower & Sennikov, 2000).

Stratigraphic horizon. Donguz Gorizont (Huene, 1940; Gower & Sennikov, 2000).

Holotype. PIN 268/2: dorsal vertebra lacking most of the neural spine.

Remarks. Huene (1940) erected “Dongusia colorata” and interpreted it as a proterosuchid. Tatarinov (1961) considered that this species was an erythrosuchid and even cogeneric with Erythrosuchus. Subsequent authors have regarded “Dongusia” and “Dongusia colorata” as nomina dubia (Young, 1964; Charig & Reig, 1970; Charig & Sues, 1976; Ezcurra, Butler & Gower, 2013) and some of them proposed that PIN 268/2 belongs to a rauisuchid archosaur (Charig & Reig, 1970; Sennikov in Gower & Sennikov, 2000). It is agreed here that this genus and species are nomina dubia, but the taxon was included in the taxonomic sample of the present analysis in order to test the original proposed proterosuchian affinities of the type specimen.

Uralosaurus magnus (Ochev, 1980)

Age. Anisian, early Middle Triassic (Tverdokhlebov et al., 2003; Lucas, 2010).

Locality. Karagachka locality, Sol’-Iletsk district, Orenburg Province, Russia (Ochev, 1980; Gower & Sennikov, 2000).

Stratigraphic horizon. Donguz Gorizont (Ochev, 1980; Gower & Sennikov, 2000).

Holotype. PIN 2973/70: left pterygoid.

Paratypes. PIN 2973/71: right dentary; PIN 2973/72–79: seven teeth.

Emended diagnosis. Uralosaurus magnus is a large archosauriform that can tentatively be distinguished from other archosauromorphs by the following combination of features (autapomorphy indicated with an asterisk): dentary strongly dorsally curved, with eight tooth positions; a pair of blind and relatively shallow fossae placed immediately posterior to the most posterior tooth socket*; and thecodont tooth implantation.

Remarks. Ochev (1980) erected the new species Erythrosuchus magnus from the Anisian of Russia based on a left pterygoid (PIN 2973/70). Ochev (1980) also referred to this species a right dentary (PIN 2973/71) and some isolated teeth from the type locality (PIN 2973/72–79), and tentatively referred some presacral vertebrae from other localities (Ochev, 1980; Gower & Sennikov, 2000). Subsequently, Sennikov (1995b) erected the new genus Uralosaurus and assigned Erythrosuchus magnus to it, resulting in the new combination Uralosaurus magnus. Sennikov (1995b) also referred to Uralosaurus magnus some cranial remains and caudal vertebrae from different localities to that of the holotype. All these elements possess an erythrosuchid-like morphology (Gower & Sennikov, 2000), but there are other species in the same stratigraphic horizon that may possess a similar morphology (e.g., Dongusuchus efremovi). Therefore, only the pterygoid, dentary, and teeth from the same locality are considered here the hypodigm of Uralosaurus magnus. This hypothesis will be tested in the phylogenetic analysis and by subsequent discoveries of associated bones of Uralosaurus magnus from the Donguz Gorizont. The overall morphology of the bones and teeth of Uralosaurus magnus (PIN 2973/70–79) is very similar to that of other erythrosuchids (e.g., Erythrosuchus africanus) and it is difficult to find characters that may support the recognition of Uralosaurus magnus as a distinct taxon. There is a pair of blind and relatively shallow fossae placed immediately posterior to the final tooth socket along the alveolar margin of the referred dentary of Uralosaurus magnus (PIN 2973/71). These fossae are too shallow to have housed teeth and are not present in any other archosauriform of which I am aware. This condition may be related with the lower tooth count (eight alveoli) of PIN 2973/71 in comparison to the dentaries of other erythrosuchids (14–16 tooth positions in Shansisuchus shansisuchus: Young, 1964; 13–14 in Garjainia prima: PIN 2394/5-8, 5-9, 951/30; 14 in Garjainia madiba: NMQR 3051; ≥12 in Erythrosuchus africanus: BP/1/3893). It is not possible to determine here if these fossae are pathological (e.g., as a result of alveolar remodeling), but, together with the very low tooth count, they may support the distinction of Uralosaurus magnus from other erythrosuchid species.

The holotype of Uralosaurus magnus and the entire hypodigm were scored as different terminals in two alternative phylogenetic analyses because of the ambiguous referral of the right dentary and teeth due to the lack of comparable homologous characters with the holotype.

Vancleavea campi Long & Murry, 1995

Age. ?Carnian–Rhaetian, Late Triassic (Hunt, Lucas & Spielmann, 2005; Nesbitt, 2011; Ramezani, Fastovsky & Bowring, 2014).

Localities. PFV 124, Petrified Forest National Park, Apache County, Arizona, USA (type locality). Several other localities in Arizona and Utah and New Mexico and Texas (Long & Murry, 1995; Hunt et al., 2002; Parker & Barton, 2008; Nesbitt et al., 2009; Nesbitt, 2011).

Stratigraphic horizons. Blue Mesa, Mesa Redondo, Monitor Butte, Sonsela, Petrified Forest, Owl Rock and “Siltstone” members of the Chinle, Bull Canyon, Redonda and Tecovas formations (Long & Murry, 1995; Hunt et al., 2002; Parker & Barton, 2008; Nesbitt et al., 2009; Nesbitt, 2011).

Holotype. PEFO 2427: incomplete postcranial skeleton.

Referred material. GR 138: complete skeleton; GR 139: partial disarticulated skeleton; and more fragmentary specimens housed at the PEFO, MNA, UCMP and UMMP palaeontological collections (Long & Murry, 1995; Hunt et al., 2002; Hunt, Lucas & Spielmann, 2005; Parker & Barton, 2008; Nesbitt et al., 2009).

Diagnosis. Nesbitt et al. (2009: 816) dignosed Vancleavea campi on the basis of the following combination of autapomorphies: absence of supratemporal and antorbital fenestrae; large caniniform, and recurved and serrated teeth in each tooth-bearing bone; well-defined depression on the lateral surface of the dentary for the maxillary caniniform tooth; neomorph bone separating the nasals; absence of a lacrimal; ilium lacking an anterior process and bearing a highly reduced posterior process; exoccipitals do not participate in the formation of the occipital condyle; dorsal centra with two paramedian ventral keels and neural spines with dorsal notches at the anterior and posterior ends; five unique osteoderm morphologies, namely teardrop-shaped ventral cervical region osteoderms, diamond-shaped osteoderms with midline keels on the ventral portion of the body, diamond-shaped osteoderms with a pointed anterior process on the lateral sides of the dorsal and caudal regions of the body, thin, mediolaterally compressed appendicular osteoderms, and large, vertically projecting, laterally compressed osteoderms dorsal to the neural spines of the caudal vertebrae.

Remarks. See comments in Nesbitt (2011: 17).

Asperoris mnyama Nesbitt, Butler & Gower, 2013

Age. Late Anisian, early Middle Triassic (Nesbitt et al., 2010).

Locality. U9/1 locality, drainage of the Hita River between the Njalila and Hiasi rivers (exact locality unknown), Songea district, southwestern Tanzania (Nesbitt, Butler & Gower, 2013).

Stratigraphic horizon. Lifua Member, Manda beds, Ruhuhu Basin (Nesbitt, Butler & Gower, 2013).

Holotype. NHMUK PV R36615: well-preserved incomplete skull including much of the right maxilla, nearly complete right premaxilla, much of the right nasal, ventral process of the postorbital, right prefrontal, right frontal, right parietal, much of right postfrontal, and other unidentified skull fragments (Nesbitt, Butler & Gower, 2013).

Diagnosis. Asperoris mnyama has the following unique combination of cranial character-states (autapomorphy indicated with an asterisk): highly sculptured cranial elements including the premaxilla, maxilla, nasal, prefrontal, frontal, postfrontal, and parietal, and highly sculptured, dorsoventrally deep orbital margin of the frontal*; posterodorsal process of the premaxilla fits into a distinct slot into the ventral process of the nasal; robust anteromedially directed palatal process of the maxilla; thecodont dentition; absence of an antorbital fossa on the maxilla anterior and ventral to the antorbital fenestra; dorsoventrally shallow antorbital fenestra; dorsoventrally thick skull roof; absence of a parietal foramen or fossa; and possible presence of a postparietal element (Nesbitt, Butler & Gower, 2013).

Remarks. Asperoris mnyama was recently named and described by Nesbitt, Butler & Gower (2013). These authors found this species as an archosauriform more crownward than Proterosuchus fergusi, but more basal than Euparkeria capensis, phytosaurus and archosaurs.

Euparkeria capensis Broom, 1913

Age. Early Anisian, early Middle Triassic (Rubidge, 2005).

Locality. Site along a road between Aliwal North and Lady Grey (exact location of the site unknown), Eastern Cape Province, South Africa (Dilkes, 1998).

Stratigraphic horizon. Burgersdorp Formation, Cynognathus AZ subzone B, Tarkastad Subgroup, Beaufort Group, Karoo Supergroup, Karoo Basin (Dilkes, 1998).

Holotype. SAM-PK-5867: partial skeleton including a nicely preserved skull and lower jaw.

Referred material. At least 11 individuals housed in the AMNH, GPIT, SAM-PK and UMZC collections (Sookias & Butler, 2013; see Ewer, 1965).

Diagnosis. No formal diagnoses of the genus Euparkeria and the species Euparkeria capensis have ever been provided (Sookias & Butler, 2013), but a detailed revision of the species is currently in preparation and will provide a diagnosis (R Sookias & R Butler, pers. comm., 2015).

Remarks. See comments in Nesbitt et al. (2009: 851).

Dorosuchus neoetus Sennikov, 1989a

Age. Anisian, early Middle Triassic (Tverdokhlebov et al., 2003; Lucas, 2010).

Localities. Berdyanka I (type and paratype locality) and Donguz I (PIN 952/200 locality) localities, Sol’Iletsk district, Orenburg Province, Russia (Sennikov, 1989a; Sennikov, 1989b).

Stratigraphic horizon. Donguz Gorizont (Sennikov, 1989a; Sennikov, 1989b).

Holotype. PIN 1579/61: mostly complete right ilium, complete right femur and complete right tibia, pertaining to a single individual and found in articulation. PIN 1579/67: a fragment of bone identified previously as the distal tip of an ischium (Sennikov, 1989a; Sennikov, 1989b) was reidentified by Sookias et al. (2014a) as the distal part of the postacetabular process of the right ilium of PIN 1579/61 and thus is also considered part of the holotype.

Paratypes. PIN 1579/62: mostly complete braincase; PIN 1579/63: single sacral vertebra lacking the neural spine, in articulation with partial sacral ribs; PIN 1579/64: two articulated proximal caudal vertebrae; PIN 1579/65: single, smaller, caudal vertebral centrum; PIN 1579/66: slightly damaged left and right ilia (smaller than the holotype); PIN 1579/68: single phalanx; PIN 952/200: partial left ilium of a similar size to those of PIN 1579/66 (Sennikov, 1989a; Sennikov, 1989b; Sookias et al., 2014a).

Diagnosis. Sookias et al. (2014a: 5, 6) diagnosed Dorosuchus neoetus as a relatively small (maximum known femur length 98.3 mm) non-crown archosauriform distinguishable on the basis of the following autapomorphies: preacetabular process of the ilium relatively short in comparison with length of pubic peduncle and distally rounded in lateral view; proximal end of tibia 33% wider dorsoventrally than distal end and approximately 70% wider than midshaft, with very little expansion of proximal end ventral to shaft (ratio of dorsal expansion of proximal end to ventral expansion: 0.22), such that dorsal margin of shaft is straight to convex and ventral margin concave in lateral view; proximal projection present proximal to dorsal margin of lateral condyle of tibia in medial view; and outline of distal end of the tibia in distal view rounded, but dorsally tapered. In addition, Sookias et al. (2014a: 6) stated that the holotype (PIN 1579/61) can be also diagnosed on the basis of the following unique combination of characters: preacetabular process of ilium short and rounded distally in lateral view; pronounced supraacetabular crest on ilium; pronounced striations on iliac blade; gently sigmoid femur; and clearly developed attachment ridge for M. caudifemoralis longus (=fourth trochanter) on femur. The paratype braincase (PIN 1579/62) is diagnosable based on the following unique combination of characters: subvertically orientated parabasisphenoid; low ridge on anterior inferior process of prootic below foramen for trigeminal nerve; and ventral ramus of opisthotic only weakly expanded laterally.

Remarks. The hypodigm of Dorosuchus neoetus is composed of the holotype right ilium, femur, and tibia (PIN 1579/61, 67), which were found in articulation and thus represent a single individual, and several paratypes (PIN 1579/62–66, 68) that were collected from the same block as the holotype (Sennikov, 1989a; Sennikov, 1989b; Sookias et al., 2014a). In addition, an isolated partial left ilium (PIN 952/200) is also considered part of the paratype because is very similar in morphology to that of PIN 1579/66 (Sookias et al., 2014a). Sookias et al. (2014a) tentatively accepted the referral of a partial left pterygoid (PIN 1579/69) and a hemimandible (PIN 1579/70) to Dorosuchus neoetus by Sennikov (1995b) and Sennikov (2008), but they tested the effect that the inclusion of these two specimens may have on the phylogenetic relationships of the animal by conducting six alternative phylogenetic analyses. These two referred specimens were found in a different block from that of the type series (Sookias et al., 2014a) and the presence of other basal archosauriforms that could possess a morphologically similar pterygoid and hemimandible in the same horizon as Dorosuchus neoetus (i.e., Dongusuchus efremovi, “Dongusia colorata”) means that this referral should be treated with caution. As a result, I decided here not to include the referred specimens of Dorosuchus neoetus when scorings this taxon; instead, Dorosuchus neoetus is scored based on the type series only.

Sennikov (1989a) referred Dorosuchus neoetus to Euparkeriidae based on its general similar morphology to Euparkeria capensis. Gower & Sennikov (2000) raised doubts about the assignment of Dorosuchus neoetus to Euparkeriidae and Sookias & Butler (2013) listed this species as a possible euparkeriid. Sookias et al. (2014a) redescribed in detail the hypodigm of Dorosuchus neoetus and included this species in a quantitative phylogenetic analysis for the first time. These authors found Dorosuchus neoetus as the sister-taxon of Phytosauria + Archosauria, immediately crownwards of Euparkeria capensis.

Yarasuchus deccanensis Sen, 2005

Age. Anisian, early Middle Triassic (Lucas, 2010).

Locality. Bhimaram village, Pranhita-Godavari Valley, Adilabad district, Andhra Pradesh, India (Sen, 2005).

Stratigraphic horizon. Yerrapalli Formation (Sen, 2005).

Holotype. ISI R 334/9–14: cervical vertebrae; ISI R334/36: first sacral; ISI R 334/37: second sacral; ISI R 334/56: right ilium; ISI R 334/63: left pubis.

Referred material. Multiple specimens collected from the same locality, listed by Sen (2005: 185, 186), and housed in the ISI palaeontological collection.

Diagnosis. Sen (2005: 185) diagnosed Yarasuchus deccanensis as a small (total length approximately 2 meters), long-necked, gracile animal, with a small skull relative to presacral length; elongated cervical vertebrae, cervical neural spine with coarsely crenulated dorsal margin; neural spines high throughout the vertebral column; pectoral girdle delicately built with small coracoid and scapula constricted near the glenoid; and osteoderms coarsely sculptured.

Remarks. Yarasuchus deccanensis was erected by Sen (2005) and assigned to the rauisuchian family Prestosuchidae. Brusatte et al. (2010) included this species for the first time in a phylogenetic analysis and recovered it as the most basal member of Poposauroidea. More recently, the hypodigm of this species was suggested to be a chimera composed of rauisuchian archosaur and prolacertiform bones (Desojo in Lautenschlager & Desojo, 2011; Nesbitt et al., 2013). After a first-hand study of the entire hypodigm of Yarasuchus deccanensis, it was not possible to recognize archosaur or putative prolacertiform apomorphies in the preserved bones (with the exception of the highly homoplastic elongated cervical vertebrae). For example, the maxilla possesses a thecodont tooth implantation and the border of the antorbital fenestra on the horizontal process lacks an antorbital fossa (ISI R334/2), and the femur possesses a fourth trochanter but lacks a posteromedial tuberosity on the proximal end (ISI R334/67), contrasting with the conditions in prolacertiforms (e.g., Protorosaurus speneri, tanystropheids, Prolacerta broomi) and suchian archosaurs (e.g., Arizonasaurus babbitti, Prestosuchus chiniquensis). The cervical vertebrae of Yarasuchus deccanensis are clearly distinct from those of Pamelaria dolichotrachela, as originally recognized by Sen (2005), differing from this allokotosaurian in the presence of a median ventral keel on the centrum, epipophysis on the postzygapophysis, and considerably taller neural spines with an anterior overhang and a rugose distal transverse thickening of the distal end. Therefore, the hypodigm of Yarasuchus deccanensis is not composed of bones that belong to prolacertiforms, rauisuchians or a combination of both, but the anatomy of the species is congruent with that expected for a non-archosaurian archosauriform. As a result, the original hypodigm of the species is retained here and scored as a single terminal.

Dongusuchus efremovi Sennikov, 1988b

Age. Anisian, early Middle Triassic (Tverdokhlebov et al., 2003; Lucas, 2010).

Locality. Donguz I site, about 1 km NW of Perovka village, along the right bank of the Donguz River (within the drainage basin of the Ural River), Sol’-Iletsk district, Orenburg Province, Russia (Sennikov, 1988b; Niedźwiedzki, Sennikov & Brusatte, 2014).

Stratigraphic horizon. Donguz Gorizont (Sennikov, 1988b; Niedźwiedzki, Sennikov & Brusatte, 2014).

Holotype. PIN 952/15-1: left femur.

Referred material. PIN 952/15-2–15-5: four femora.

Diagnosis. Niedźwiedzki, Sennikov & Brusatte (2014: 9) diagnosed Dongusuchus efremovi as a non-archosaurian archosauriform with the following autapomorphies: remarkably gracile femur with a ratio of proximodistal length to minimum midshaft diameter in posteromedial or anterolateral view of >13.0; very deep and curved groove on the proximal surface of the femoral head; and a low, nearly flat and medially displaced posteromedial tuber on the proximal portion of the femur in proximal view, devoid of anteromedial tuber.

This combination of apomorphies is also present in the femora of the hypodigm of Yarasuchus deccanensis and it is not possible to distinguish the two species from each other. Nevertheless, the equivocally referred specimens of Dongusuchus efremovi (sensu Niedźwiedzki, Sennikov & Brusatte, 2014) differ from the same elements in Yarasuchus deccanensis. As a result, Dongusuchus efremovi may potentially be a senior synonym of Yarasuchus deccanensis, but the taxonomic affinities of the equivocally referred specimens of the former species should be resolved in order to shed light on the taxonomy of both taxa.

Remarks. Sennikov (1988b) erected the new genus and species Dongusuchus efremovi based on an isolated complete left femur (PIN 952/15-1). Several additional postcranial bones have been subsequently referred to this species (Sennikov, 1988a; Sennikov, 1988b; Sennikov, 1990; Sennikov, 1995b). Recently, Niedźwiedzki, Sennikov & Brusatte (2014) considered only four femora demonstrably referred specimens (PIN 952/15-2–5), and the other previously referred specimens that cannot be compared directly with the holotype because they lack overlapping features were considered equivocally referred specimens. Niedźwiedzki, Sennikov & Brusatte (2014) conducted alternative phylogenetic analyses testing the effect that the inclusion of the equivocally referred specimens has on the phylogenetic position of the species. The results of these analyses showed that these equivocally referred specimens do not affect the position of Dongusuchus efremovi (Niedźwiedzki, Sennikov & Brusatte, 2014). In the present phylogenetic analysis the scorings for this species were based only upon the holotype and demonstrably referred specimens. As for Dorosuchus neoetus, the presence of other basal archosauriforms in the same horizon as Dongusuchus efremovi (e.g., Dorosuchus neoetus, “Dongusia colorata”) means that caution is required in the referral of bones that do not overlap in morphology with the holotype (e.g., cervical vertebrae, forelimb elements).

Proterochampsa barrionuevoi Reig, 1959

Age. Late Carnian–earliest Norian, early Late Triassic (Rogers et al., 1993; Furin et al., 2006; Martínez et al., 2011).

Localities. Several localities of the Hoyada de Ischigualasto, Ischigualasto Provincial Park, San Juan Province, Argentina (Reig, 1959; Sill, 1967; Martínez et al., 2012).

Stratigraphic horizon. Upper La Peña, Cancha de Bochas and lower Valle de la Luna members of the Ischigualasto Formation, Ischigualasto-Villa Unión Basin (Martínez et al., 2012).

Holotype. PVL 2063: skull with lower jaw and five presacral vertebrae. Dilkes & Arcucci (2012) reported that the vertebrae cannot currently be located.

Referred material. MACN-Pv 18165: partial skull with one hemimandible; MCZ 3408: complete skull with lower jaw and 13 articulated presacral vertebrae and ribs; PVL 2057: partial skull with lower jaw; PVL 2058: skull with lower jaw; PVL 2061: partial skull lacking snout, palate, braincase and lower jaw; PVL 3434: skull lacking premaxillae, most of the temporal region, braincase, and lower jaw; PVSJ 77: complete skull with one articulated and one separate hemimandible.

Diagnosis. Trotteyn, Arcucci & Raugust (2013: 62, 63) diagnosed Proterochampsa barrionuevoi by the following autapomorphies: dermal sculpture consisting of highly nodular protuberances and prominent ridges with smaller periodic nodular growths along their length; antorbital fossa restricted to elongate depression on the maxilla at anterior end of the antorbital fenestra; lateral expansion of premaxillae anterior to contact between premaxilla and maxillae along ventral margin of skull; no fossa along postorbital, squamosal and parietal borders of supratemporal fenestra; large anteriorly curved spine on quadratojugal; exclusion of jugal from suborbital fenestra by contact of maxilla and ectopterygoid; basal tubera of parabasisphenoid faces ventrally rather tan ventrolaterally and projects laterally beyond the basipterygoid process; and a ventrally projecting lamina on the angular. Proterochampsa barrionuevoi is also characterized by the absence of a retroarticular process, occipital wing of the parietal diverges from the midline axis of the skull at an angle of approximately 60°, and dorsoventrally flattened skull with antorbital fenestrae, orbits and supratemporal fenestrae dorsal in orientation, in addition to the dorsal position of the external nares.

Remarks. Proterochampsa barrionuevoi is one of the best-known non-archosaurian archosauriforms (Sill, 1967; Trotteyn, 2011a; Dilkes & Arcucci, 2012), but it was included in quatitative phylogenetic analyses as an independent terminal (and not part of a suprageneric Proterochampsidae) only recently (Dilkes & Arcucci, 2012; Trotteyn & Ezcurra, 2014; Ezcurra, Desojo & Rauhut, 2015). Proterochampsa barrionuevoi has been recovered as the most basal member of the South American proterochampsid radiation and, as a result, it is a key taxon to optimize the ancestral character-states of the group.

Proterochampsa nodosa Barberena, 1982

Age. Late Carnian–earliest Norian, early Late Triassic (Rogers et al., 1993; Langer, 2005; Furin et al., 2006; Martínez et al., 2011).

Locality. Sesmaria do Pinhal, Rio Grande do Sul State, Brazil (Barberena, 1982; Langer et al., 2007).

Stratigraphic horizon. Santa Maria Sequence 2, Hyperodapedon AZ, Rosário do Sul Group, Paraná Basin (Barberena, 1982; Langer et al., 2007).

Holotype. MCP 1694 PV: partial skull and lower jaw.

Emended diagnosis. Trotteyn, Arcucci & Raugust (2013: 66) diagnosed Proterochampsa nodosa as a proterochampsian with a skull that is proportionally similar to the skull of Proterochampsa barrionuevoi; occipital region proportionally higher than in Proterochampsa barrionuevoi; quadrate forms an angle of 90°with the squamosal; ornamented skull with fewer nodules than in Proterochampsa barrionuevoi; elongated nares; and frontal more elongated and whose anterior half is flatter than in Proterochampsa barrionuevoi.

Remarks. This species is known from a rather complete skull that it is very similar to its Argentinean congeneric species Proterochampsa barrionuevoi (Barberena, 1982). Proterochampsa nodosa has been included only recently in a quantitative phylogenetic analysis and its result bolstered the monophyly of Proterochampsa (Ezcurra, Desojo & Rauhut, 2015).

Tropidosuchus romeri Arcucci, 1990

Age. Early Carnian, early Late Triassic (Marsicano et al., 2015).

Locality. Los Chañares type locality, near the mouth of Chañares River, Talampaya National Park, La Rioja Province, Argentina.

Stratigraphic horizon. Lower member of the Chañares Formation (sensu Fiorelli et al., 2013), Ischigualasto-Villa Unión Basin.

Holotype. PVL 4601: articulated skeleton lacking the manus.

Referred material. MCZ 9482: partial skeleton associated with a specimen of Gracilisuchus stipanicicorum; PULR unnumbered: partial postcranium associated with the holotypes of “Lagosuchus talampayensis” (PULR 09) and Gracilisuchus stipanicicorum (PULR 08); PVL 4602: anterior portion of the skull, lower jaw, articulated presacral column, partial pelvic girdle, and both femora, tibiae, and fibulae; PVL 4603: skull and lower jaw with presacral column, partial pelvic girdle, seven articulated caudal vertebrae; and several disarticulated osteoderms; PVL 4604: fairly complete skeleton; PVL 4605: skull and lower jaw with articulated presacral vertebral column, seven proximal caudal vertebrae, seven distal caudal vertebrae, articulated left scapula, coracoid, and humerus, pelvic girdle, both femora, tibiae, and fibulae, articulated left distal tarsals and pes; PVL 4606: skull and lower jaw, articulated presacral vertebral series, sacrum and pelvic girdle, left femur, tibia, fibula, tarsus, and pes; and PVL 4624: left femur, tibia and fibula, and one metatarsal (modified from Trotteyn, Arcucci & Raugust, 2013).

Diagnosis. Trotteyn, Arcucci & Raugust (2013: 78, 79) diagnosed Tropidosuchus romeri as an archosauriform with triangular skull in dorsal view; proportionally large orbits; curved premaxilla at the distal tip; nearly vertical quadrate; occipital crest well developed; ornamentation of the skull in longitudinal crests with different disposition; pterygoid with posterior edges of the wings straight and thickened; internal edge of pterygoid with denticles in alveoli, forming a ‘V’-shaped structure opened posteriorly; vertebral cervico-dorsal zonation more marked; shoulder girdle with rod-like clavicle and interclavicle; ulna without olecranon; femur as long as the tibia; tibia with distal articulation not transverse or longitudinal; and large osteoderms, in one row and one per vertebra, with a well-developed dorsal laminar crest extended axially.

Remarks. See comments in Nesbitt et al. (2009).

Cerritosaurus binsfeldi Price, 1946

Age. Late Carnian–earliest Norian, early Late Triassic (Rogers et al., 1993; Langer, 2005; Furin et al., 2006; Martínez et al., 2011).

Locality. Sanga do Mato, Rio Grande do Sul State, Brazil (Price, 1946; Langer et al., 2007).

Stratigraphic horizon. Santa Maria Sequence 2, Hyperodapedon AZ, Rosário do Sul Group, Paraná Basin (Price, 1946; Langer et al., 2007).

Holotype. CA s/n: complete skull and lower jaw articulated with the first 15 presacral vertebrae, ribs, and osteoderms, and a humerus.

Diagnosis. Trotteyn, Arcucci & Raugust (2013: 68) diagnosed Cerritosaurus binsfeldi as a proterochampsian with a dorsoventrally compressed and ornamented skull roof; ornamentation of the skull roof is more visible in the posterior region of the nasal to the parietal; supratemporal fenestra small and similar in length to the antorbital fenestra; external naris small and oval, surrounded by slit-like depressions on the premaxilla and nasal; and oval osteoderms located on the neural spines, forming one longitudinal row.

Remarks. The holotype and single known specimen of Cerritosaurus binsfeldi is preserved in a concretionary matrix and some bones are strongly taphonomically altered, limiting available information on the palate, braincase, and postcranium. This species was included for the first time in a quantitative phylogenetic analysis by Dilkes & Arcucci (2012) and recovered as the sister-taxon of the early Late Triassic Argentinean proteochampsids Tropidosuchus romeri, Chanaresuchus bonapartei and Gualosuchus reigi. A congruent phylogenetic position was found by subsequent analyses (Trotteyn & Ezcurra, 2014; Ezcurra, Desojo & Rauhut, 2015).

Gualosuchus reigi Reig, 1971a

Age. Early Carnian, early Late Triassic (Marsicano et al., 2015).

Locality. Los Chañares type locality, near the mouth of Chañares River, Talampaya National Park, La Rioja Province, Argentina.

Stratigraphic horizon. Lower member of the Chañares Formation (sensu Fiorelli et al., 2013), Ischigualasto-Villa Unión Basin.

Holotype. PULR 01: right half side of the skull without braincase, partial right hemimandible, some presacral vertebrae, left scapula, coracoid, femur, tibia, and fibula.

Referred material. PVL 4576: complete skull and lower jaw, presacral vertebral column, sacrum articulated with the pelvis, complete forelimb without manus, scapula, coracoid, femur, tibia, and fibula (Dilkes & Arcucci, 2012).

Diagnosis. Trotteyn, Arcucci & Raugust (2013: 76) diagnosed Gualosuchus reigi as an archosauriform with skull posteriorly taller and narrower than Proterochampsa, Chanaresuchus bonapartei or Tropidosuchus romeri; oval orbits with dorsoventral major axis; parietals diverge posteriorly; supratemporal fenestra narrow and axially elongated; vertebral zonation noticeable, with cervico-dorsal differentiation; humerus robust and shorter than Chanaresuchus bonapartei; femur shorter than the tibia and fibula, and shorter than Chanaresuchus bonapartei; and dorsal osteoderms in one row over the neural spines, one per vertebra, oval in shape and with a smooth surface.

Remarks. Gualosuchus reigi has been identified since its original description as closely related to its probably sympatric species Chanaresuchus bonapartei (Romer, 1971a; Trotteyn, Arcucci & Raugust, 2013). Recent quantitative phylogenetic analyses have bolstered this hypothesis (Dilkes & Arcucci, 2012; Trotteyn & Ezcurra, 2014; Ezcurra, Desojo & Rauhut, 2015).

Chanaresuchus bonapartei Reig, 1971a

Age. Early Carnian, early Late Triassic (Marsicano et al., 2015).

Locality. Los Chañares type locality, near the mouth of Chañares River, Talampaya National Park, La Rioja Province, Argentina.

Stratigraphic horizon. Lower member of the Chañares Formation (sensu Fiorelli et al., 2013), Ischigualasto-Villa Unión Basin.

Holotype. PULR 07: skull and lower jaw, cervical and dorsal vertebrae and ribs, pelvic girdle, and left femur, tibia and fibula.

Referred material. MCZ 4035: articulated vertebral series up to the third caudal vertebra, scapular and pelvic girdle, and partial forelimbs and hindlimbs; MCZ 4036: partial and in part poorly preserved cranial remains and postcranial materials of two additional individuals; MCZ 4037: skull and presacral column; MCZ 4038: slab containing remains of a nearly completely disarticulated skeleton; MCZ 4039: well-preserved left half of a small skull;PVL 4575: partial skull, right hemimandible, cervical, dorsal, sacral, and caudal vertebrae, right scapula and coracoid, right humerus, radius, and ulna, pelvic girdle, and both femora and tibiae; PVL 4586: skull and lower jaw; PVL 4647: partial skull; PVL 4676: partial skull; and PVL 6244: partial postcranium.

Diagnosis. Trotteyn, Arcucci & Raugust (2013: 70) diagnosed Chanaresuchus bonapartei as a proterochampsian with the following unique combination of character-states (autapomorphy indicated with an asterisk): skull long and low, broad posteriorly; slit-like external nares placed close together dorsally some distance back from the tip of the snout; antorbital fenestra small; parietals swing sharply outwards posteriorly above supratemporal fenestrae, towards their meeting with the squamosals; suspensorium far back of occiput with lateral fenestra elongated anteroposteriorly; elongate choanae partially covered below by a secondary palate; long and narrow interpterygoid vacuity, exposing a slender parasphenoid rostrum; and osteoderms wedge-shaped, narrow anteriorly and broader posteriorly*.

Remarks. Chanaresuchus bonapartei is the best-known proterochampsid based on both quantity and quality of the preserved specimens. The anatomy of the species was described by Romer (1971a) and Romer (1972b). Chanaresuchus bonapartei has been used to represent proterochampsid morphology in previous phylogenetic analyses (e.g., Ezcurra, Lecuona & Martinelli, 2010; Desojo, Ezcurra & Schultz, 2011; Nesbitt, 2011; Butler et al., 2014a).

Pseudochampsa ischigualastensis (Trotteyn, Martínez & Alcober, 2012)

Age. Late Carnian–earliest Norian, early Late Triassic (Rogers et al., 1993; Furin et al., 2006; Martínez et al., 2011).

Locality. Valle Pintado, Ischigualasto Provincial Park, San Juan Province, Argentina (Trotteyn, Martínez & Alcober, 2012).

Stratigraphic horizon. Cancha de Bochas Member, Ischigualasto Formation, Ischigualasto-Villa Unión Basin (Trotteyn, Martínez & Alcober, 2012).

Holotype. PVSJ 567: fairly complete, articulated skeleton including skull with fully occluded mandible, complete vertebral series lacking the distal half of the tail, several cervical and dorsal ribs, some haemal arches, some gastralia, pectoral girdle, both partial humeri, partial pelvic girdle, both femora, tibiae, fibulae, tarsals, and pes.

Diagnosis. Pseudochampsa ischigualastensis is distinguished from other proterochampsids, including Chanaresuchus bonapartei, on the basis of the following unique combination of character-states: basicranium transversely broad (basal tubera width/parabasisphenoidal complex axial length ratio = 0.31) and with transversely oriented basal tubera; paroccipital processes with dorsoventrally expanded distal end; lower jaws without retroarticular process; caudal vertebrae with a median longitudinal groove on the ventral surface of the centrum, and pre- and postzygapophyses strongly divergent from the median line; astragalus lacking foramina on the posterior groove; and osteoderms with an ornamentation consisting only of a longitudinal groove (Trotteyn & Ezcurra, 2014: 4).

Remarks. Trotteyn, Martínez & Alcober (2012) erected and briefly described the new species “Chanaresuchus” ischigualastensis on the basis of a fairly complete, articulated skeleton. Trotteyn & Haro (2012) described in detail the braincase of the species and found a monophyletic Chanaresuchus in a quantitative phylogenetic analysis that sampled only braincase characters. Trotteyn & Ezcurra (2014) described in detail the entire anatomy of “Chanaresuchus” ischigualastensis and included it for the first time in a phylogenetic analysis sampling the entire anatomy of the terminals. These authors did not find unequivocal evidence for the monophyly of Chanaresuchus and, as a result, erected the new genus Pseudochampsa for the species, resulting in the new combination Pseudochampsa ischigualastensis. More recently, Ezcurra, Desojo & Rauhut (2015) found Pseudochampsa ischigualastensis and Gualosuchus reigi in a polytomy together with a clade composed of Chanaresuchus bonapartei and Rhadinosuchus gracilis, thus supporting the non-monophyly of the genus Chanaresuchus (sensu Trotteyn, Martínez & Alcober, 2012).

Rhadinosuchus gracilis Huene, 1938

Age. Late Carnian–earliest Norian, early Late Triassic (Rogers et al., 1993; Langer, 2005; Furin et al., 2006; Martínez et al., 2011).

Locality. Quarry 17 of the Sanga 6 or ‘Zahn Sanga’, São José, Rio Grande so Sul State, southern Brazil (Ezcurra, Desojo & Rauhut, 2015).

Stratigraphic horizon. Alemoa Member, Santa Maria Sequence 2, Hyperodapedon AZ, Rosário do Sul Group, Paraná Basin (Ezcurra, Desojo & Rauhut, 2015).

Holotype. BSPG AS XXV 50, 51: partial skull and postcranium originally preserved in two blocks of red mudstone. The skull includes both premaxillae and dentaries, right maxilla, nasal, lacrimal and anterior tip of frontal, left jugal, quadratojugal, opisthotic, exoccipital, prootic, and splenial. The postcranium is represented by a posterior cervical centrum, two partial cervical ribs, several gastralia, two dorsal osteoderms, and a probable left metatarsal II. In addition, a possible partial neural arch of the axis and an indeterminate bone are preserved in the main block (Ezcurra, Desojo & Rauhut, 2015).

Diagnosis. Rhadinosuchus gracilis is a proterochampsian distinguished from other basal archosauriforms by the following combination of character-states (autapomorphy indicated with an asterisk): maxilla with a dorsoventrally low antorbital fossa on the horizontal process; nasal with an anteroposteriorly elongated narial fossa and strongly ornamented dorsal surface composed of mainly longitudinally oriented ridges; lacrimal with a very well anteroposteriorly developed antorbital fossa on the ventral process; and dentary with a large, anterodorsally opening foramen on the anterior surface*; and more than 22 dentary tooth positions (Ezcurra, Desojo & Rauhut, 2015: 394).

Remarks. Rhadinosuchus gracilis was the first described proterochampsid but was originally interpreted as a pseudosuchian by Huene (1938). Subsequently, different authors proposed several alternative phylogenetic positions for the species (e.g., Romer, 1945; Romer, 1956; Romer, 1966; Hoffstetter, 1955; Kuhn, 1961; Reig, 1961; Reig, 1970; Bonaparte, 1970), but, in most cases, they agreed that it was an archosaur closely related to Cerritosaurus binsfeldi (see Ezcurra, Desojo & Rauhut, 2015). Indeed, Hoffstetter (1955) considered that Rhadinosuchus gracilis was a probable subjective senior synonym of Cerritosaurus binsfeldi, but this hypothesis was dismissed by other authors (Huene, 1956; Reig, 1970; Bonaparte, 1971) and more recently refutted by Ezcurra, Desojo & Rauhut (2015). After the original descriptions of Chanaresuchus bonapartei and Gualosuchus reigi (Romer, 1971a; Romer, 1972b), Romer (1972a) proposed that Rhadinosuchus gracilis was a member of the family Proterochampsidae, together with the former species and Proterochampsa barrionuevoi and Cerritosaurus binsfeldi. However, this interpretation was not followed by other authors (Sill, 1974; Krebs, 1976). The idea of Rhadinosuchus gracilis as a proterochampsid was readopted by Kischlat & Schultz (1999) and Kischlat (2000), and followed by recent authors (e.g., Dilkes & Arcucci, 2012; Raugust, Lacerda & Schultz, 2013; Trotteyn, Arcucci & Raugust, 2013; Trotteyn & Ezcurra, 2014). More recently, Ezcurra, Desojo & Rauhut (2015) redescribed in detail the holotype and only known specimen of Rhadinosuchus gracilis and included it for the first time in a quantitative phylogenetic analysis. These authors found this species as more closely related to Chanaresuchus bonapartei than to other proterochampsids.

Tarjadia ruthae Arcucci & Marsicano, 1998

Age. Late Ladinian, late Middle Triassic (Marsicano et al., 2015).

Localities. Big bend of the Gualo River, near Agua Escondida (type locality), and Río Chañares, 1.5 kilometres to the north of the Chañares type locality, Talampaya National Park, La Rioja Province, Argentina (Arcucci & Marsicano, 1998).

Stratigraphic horizon. Lowermost levels (0–5 metres) of the lower member of the Chañares Formation (sensu Fiorelli et al., 2013), Ischigualasto-Villa Unión Basin.

Holotype. PULR 063: several dorsal osteoderms and partial vertebrae.

Referred material. MCZ 9319: partial skull roof, including frontals, postfrontal and parietals, supraoccipital, posterior end of the left surangular, and a fragment of dorsal osteoderm; MCZ 4076: posterior end of the right hemimandible, some vertebral fragments, and four dorsal osteoderms; and MCZ 4077: partial right femur and partial dorsal osteoderms.

Emended diagnosis. Tarjadia ruthae is a basal archosauriform that differs from other archosauromorphs by the following combination of features (autapomorphy indicated with an asterisk): dorsoventrally thick skull roof and ornamented by deep pits and grooves of random arrangment*; postfrontal bone present; supraoccipital without median vertical peg; surangular with a strongly laterally developed shelf on the dorsolateral surface of the bone; dorsal vertebrae with a moderately transversely compressed centrum at mid-length; and femur with a poorly developed fourth trochanter.

Remarks. Arcucci & Marsicano (1998) described the remains of a new putative archosaur from the Middle Triassic of Argentina that was named Tarjadia ruthae. The phylogenetic affinities of this new genus and species remained enigmatic until the description of Archeopelta arborensis from coeval beds of southern Brazil. Desojo, Ezcurra & Schultz (2011) proposed that Tarjadia ruthae and Archeopelta arborensis were closely related to each other and also to the North American Late Triassic species Doswellia kaltenbachi, as members of the family Doswelliidae. Desojo, Ezcurra & Schultz (2011) found doswelliids as non-archosaurian archosauriforms more crownwards than Proterosuchus spp., Erythrosuchus africanus, Euparkeria capensis and Chanaresuchus bonapartei.

Arcucci & Marsicano (1998) identified the preserved cranial occipital region of MCZ 9319 as composed of the supraoccipital and the paroccipital processes of the opisthotics. However, the ventral margin of this fragment of skull was originally misinterpreted as dorsal and the dorsal as ventral, and the putative opisthotics are actually the posteroventral processes of the parietals (MCZ 9319). Furthermore, an additional referred specimen of Tarjadia ruthae (MCZ 4077) was identified in the collection of the MCZ.

Archeopelta arborensis Desojo, Ezcurra & Schultz, 2011

Age. Ladinian–early Carnian, late Middle–early Late Triassic (Langer et al., 2007; Desojo, Ezcurra & Schultz, 2011; Philipp et al., 2013; Marsicano et al., 2015).

Locality. Sanga da Árvore (Baum Sanga), Xiniquá region, São Pedro do Sul, Rio Grande do Sul State, Brazil (Desojo, Ezcurra & Schultz, 2011).

Stratigraphic horizon. Santa Maria 1 Sequence, Dinodontosaurus AZ (Desojo, Ezcurra & Schultz, 2011).

Holotype. CPEZ-239a: basicranium, a series of 13 dorsal vertebrae, three dorsal neural spines, two dorsal ribs, two sacral neural arches and their ribs, two sacral or caudal centra, ten paramedian osteoderms, three lateral osteoderms, four undetermined osteoderms, proximal end of right humerus, proximal half of right ulna, right ilium, right ischium, right femur, and proximal end of right tibia (Desojo, Ezcurra & Schultz, 2011).

Diagnosis. Archeopelta arborensis is a doswelliid archosauriform distinguished from other archosauromorphs on the basis of the following combination of features (autapomorphies indicated with an asterisk): basioccipital without occipital neck separating the occipital condyle from the rest of the basicranium*; opistothical paroccipital processes with large and oval fossa on their dorsomedial corner*; suture between the parabasisphenoid and basioccipital interdigitated and V-shaped in ventral view; parabasisphenoid with minute and strongly posteriorly displaced foramina for the internal carotid artery*; dorsal centra without a transverse constriction; first primordial sacral vertebra with circular and extremely large prezygapophyses accounting for 43% of the total length of the neural arch*, and well-developed V-shaped hyposphene*; humerus with a proximomedially orientated head; ilium with base of the iliac blade medially deflected*; femur with strongly transversely expanded distal end representing approximately 150% of the transverse width of the femoral head* (Desojo, Ezcurra & Schultz, 2011: 844).

Remarks. Archeopelta arborensis was named by Desojo, Ezcurra & Schultz (2011) and its description shed new light on the taxonomic content and phylogenetic relationships of Doswelliidae. Lucas, Spielmann & Hunt (2013) suggested that the diagnostic characters of Archeopelta listed by Desojo, Ezcurra & Schultz (2011) do not distinguish the genus from Tarjadia, simply because they pertain to anatomy not known from the fragmentary type material of the latter species. Therefore, Lucas, Spielmann & Hunt (2013) proposed that Archeopelta and Archeopelta arborensis are subjective junior synonyms of Tarjadia and Tarjadia ruthae, respectively. However, this proposal is not followed here because it is based on weak grounds. The occipital surface of the supraoccipital of Tarjadia ruthae is well preserved and clearly lacks a vertical ridge (MCZ 9319), contrasting with the condition in Archeopelta arborensis (Desojo, Ezcurra & Schultz, 2011). The dorsal vertebrae of Tarjadia ruthae are transversely constricted at mid-length (PULR 063), but they are not constricted in Archeopelta arborensis (Desojo, Ezcurra & Schultz, 2011). As a result, the anatomy of Archeopelta arborensis clearly differs from that of Tarjadia ruthae, and they are considered different species. Both species has been scored as independent terminals in the present phylogenetic analysis.

Jaxtasuchus salomoni Schoch & Sues, 2014

Age. Ladinian, late Middle Triassic (Schoch & Sues, 2014).

Localities. Schumann Quarry, Vellberg (Eschenau), east of Schwäbisch Hall, Baden-Württemberg, Germany (type locality). Referred specimens also come from the Kupferzell, Wolpertshausen, Zwingelhausen, Rielingshausen, and Vellberg localities; Baden-Württemberg, Germany (Schoch & Sues, 2014).

Stratigraphic horizon. Upper member of the Lower Keuper, Erfurt Formation (Schoch & Sues, 2014).

Holotype. SMNS 91352A–C: mostly articulated skeleton lacking the skull and cervical vertebrae, recovered as several blocks. SMNS 91352A and B comprise part and counterpart of the anterior portion of the trunk including the forelimbs, and SMNS 91352C preserves the pelvic region, hindlimbs and much of the tail.

Referred material. SMNS 91083: partial, disarticulated anterior portion of a skeleton including both maxillae, probable left postorbital, anterior part of the left nasal, posterior process of left premaxilla, right pterygoid, supraoccipital, basioccipital, both exoccipitals, and left angular; seven transverse rows of osteoderms, each comprising four elements; cervical vertebrae 2–8 and cervical ribs; SMNS 91002: complete forelimb, dorsal osteoderms, and associated femur; SMNS 90500: caudal osteoderms, two vertebrae, and two ribs; SMNS 81868: one dorsal vertebra; SMNS 81906, 90046: two caudal vertebrae; SMNS 81891–81905, 90530–90539: 25 osteoderms; SMNS 90505, dorsal vertebrae, numerous osteoderms, ribs, and possible gastralia; SMNS 90067, 90068: two paramedian osteoderms; SMNS 59403: lateral caudal osteoderm.

Diagnosis. Jaxtasuchus salomoni is distinguished by the following combination of autapomorphies: maxillary teeth with tall, slightly recurved crowns that have prominent vertical ridges along labial and lingual surfaces of tooth crowns and smooth mesial and distal carinae; width of dorsal osteoderms up to 1.2–1.8 times their greatest length; ornamentation most pronounced on dorsal and caudal paramedian osteoderms, comprising prominent ridges and deep pits that are more strongly developed but fewer in number than on comparable osteoderms of Doswellia kaltenbachi; and cervical vertebrae with centra distinctly longer than high, with sixth and seventh cervicals being the longest (Schoch & Sues, 2014: 114).

Remarks. Schoch & Sues (2014) named the new doswelliid archosauriform Jaxtasuchus salomoni and this species currently represents the most completely known Middle Triassic member of the group. Jaxtasuchus salomoni was recovered as more closely related to the North American Doswellia kaltenbachi than to the South American doswelliids (Schoch & Sues, 2014).

Doswellia kaltenbachi Weems, 1980

Age. Carnian, early Late Triassic (Dilkes & Sues, 2009).

Localities. Pit dug for foundation of Doswell sewer plant, 0.4 miles northwest of the confluence of the North Anna River and the Little River, near Doswell, Hanover County, Virginia, USA (type locality); some other localities from the Taylorsville Basin listed by Weems (1980); TMM Localities 31025 and 31098, Texas, USA (Weems, 1980; Long & Murry, 1995).

Stratigraphic horizons. Poor Farm Member, Falling Creek Formation, Doswell Group, Taylorsville Basin (type horizon); and “Pre-Tecovas Horizon” of the Dockum Group, Howard County (Weems, 1980; Long & Murry, 1995).

Holotype. USNM 244214: axial skeleton from seventh cervical through fifth caudal, scattered more posterior caudals, associated ribs, partial pelvic girdle, clavicle, interclavicle, dorsal and lateral armour badly shattered except for an articulated patch from the posterior region (Weems, 1980).

Paratype. USNM 214823: postorbital portion of the skull, postdentary bones of the mandible, second through fifth cervicals, cervical ribs, nuchal armour, cervical osteoderms, and the distal end of a ?tibia (probably from the same individual as the holotype) (Weems, 1980).

Referred material. USNM 25840: two presacral vertebrae; USNM 437574: isolated right jugal; USNM 186989: one cervical vertebra, one posterior dorsal vertebra, left dentary and femur and some osteoderms; USNM 244215: anterior dorsal vertebral centrum; TMM 31025-64: seven dorsal vertebrae; TMM 31025-152: several osteoderms; TMM 31025-153: some osteoderms; TMM 31098-45: much of disarticulated dorsal armour and a single cervical vertebra (Weems, 1980; Long & Murry, 1995; Dilkes & Sues, 2009).

Diagnosis. Dilkes & Sues (2009: 59) diagnosed Doswellia kaltenbachi as an archosauromorph diapsid characterized by the following autapomorphies: elongate diapophyses of dorsal vertebrae with ventral concave and rugose surfaces for articulation with elongate capitulum of dorsal ribs; sharply angled cervical and anterior dorsal ribs; abrupt change in cross-sectional shape of rib cage from narrow to wide between anterior and posterior dorsal vertebrae; extensive series of osteoderms forming transverse rows from back of skull to at least base of tail and including at least five longitudinal rows on each side of vertebral column in posterior dorsal region; ilium with laterally deflected dorsal blade. In addition, Dilkes & Sues (2009: 59) proposed that Doswellia kaltenbachi is also distinguished by the following unique combination of features: prominent occipital peg of supraoccipital that projects over dorsal rim of foramen magnum; euryapsid construction of temporal region with enlarged jugal below supratemporal fenestra; absence of postparietals, tabulars, and postfrontals; small elliptical supratemporal fenestra that does not reach occipital margin; squamosals with posteriorly directed “horn-like” processes; elongate convex dorsal end of quadrate that fits into elongate ventral groove on squamosal; step between the flat skull roof and temporal region; absence of lateral mandibular fenestra; teeth with slender, conical crowns lacking carinae; three sacral ribs, the first derived from dorsal region; and a pair of oval articular facets at distal tips of first two caudal ribs.

Remarks. Doswellia kaltenbachi was named by Weems (1980) and currently represents the best-known doswelliid after a detailed redescription conducted by Dilkes & Sues (2009). Weems (1980) originally considered Doswellia kaltenbachi an archosaur, but that it belonged to its own family and even suborder. Weems (1980) also noted similarities between this species and aetosaurs, though he dismissed any close relationship between them. Nevertheless, Bonaparte (1982) referred Doswellia kaltenbachi to the Aetosauria. Benton & Clark (1988) included this species in a quantitative phylogenetic analysis for the first time and found it as the sister-taxon of Proterochampsidae. This result was also recovered by Dilkes & Sues (2009), but Ezcurra, Lecuona & Martinelli (2010) and Desojo, Ezcurra & Schultz (2011) found Doswellia kaltenbachi as more closely related to Archosauria than to proterochampsids.

Parasuchus hislopi (Lydekker, 1885)

Age. Late Carnian–earliest Norian, early Late Triassic (Rogers et al., 1993; Langer, 2005; Furin et al., 2006; Martínez et al., 2011).

Localities. Vicinity of Mutapuram village, Pranhita–Godavari Valley, Telangana (neotype and referred locality), and Rewa Basin Shadol District, Madhya Pradesh, India (Chatterjee, 1978; Kammerer et al., 2015).

Stratigraphic horizons. Lower Maleri (neotype and referred horizon) and Tiki formations (Chatterjee, 1978; Kammerer et al., 2015).

Neotype. ISI R42: skull with lower jaw and articulated postcranium.

Referred material. ISI R43: partial articulated skeleton lacking the forelimbs and the anterior portion of the skull (found associated with the neotype); ISI R44: partial skull (some elements of which are now lost).

Diagnosis. Kammerer et al. (2015: 7) diagnosed Parasuchus hislopi as a species of Parasuchus distinguished from Parasuchus bransoni by a relatively low narial eminence with a raised, rugose posterior margin of the naris (a ‘narial rim’); distinguished from Parasuchus angustifrons by the absence of paired depressions on the anterior portion of the nasals; and tentatively distinguished from Parasuchus magnoculus by the posterior confluence of the raised margins of the nares.

Remarks. Parasuchus hislopi was originally erected by Lydekker (1885) on the basis of a specimen that is currently considered non-diagnostic (Chatterjee, 1978). A neotype for Parasuchus hislopi (ISI R42) was subsequently designated (Chatterjee, 2001), and Chatterjee (1978) described in detail the available specimens of the species. More recently, Kammerer et al. (2015) revised the taxonomy of basal phytosaurs and concluded that Parasuchus hislopi was a valid species, unlike some previous authors (e.g., Gregory, 1962; Westphal, 1976; Hunt & Lucas, 1991; Long & Murry, 1995), and Parasuchus was a subjective senior synonym of Paleorhinus and Arganarhinus, in agreement with Lucas, Heckert & Rinehart (2007). Parasuchus hislopi has been usually considered one of the most basal known phytosaurs and this hypothesis has been recently supported by quantitative phylogenetic analyses (e.g., Kammerer et al., 2015).

Parasuchus angustifrons (Kuhn, 1936)

Age. Late Carnian, early Late Triassic (Butler et al., 2014b).

Locality. Bed 9 of the Ebrach quarry, Bamberg district, Upper Franconia region of northern Bavaria, Germany (Kuhn, 1933; Kuhn, 1936; Butler et al., 2014b).

Stratigraphic horizon. Blasensandstein of the Sandsteinkeuper, laterally equivalent to the Hassberge Formation of the Middle Keuper (Kuhn, 1933; Kuhn, 1936; Butler et al., 2014b).

Holotype. BSPG 1931 X 502: skull missing the anterior portions of the premaxillae.

Diagnosis. Parasuchus angustifrons is characterized by the following autapomorphies: stepped lateral rim of external naris that is strongly swollen and rugose at posterior end; paired depressions on the anterior portions of the nasals (immediately posterior to the external nares) and anterior portions of the frontals; foramen in ectopterygoid enlarged and subcircular in outline; suborbital foramen elongate and boomerang-shaped; and large postparietal foramen at junction between supraoccipital and parietal (Butler et al., 2014b: 8).

Remarks. Kuhn (1936) erected the new species “Francosuchus” angustifrons based on a partial, well-preserved skull. Subsequently, this species was transferred to the genus “Paleorhinus” and its taxonomic validity was questioned (Gregory, 1962; Hunt & Lucas, 1991; Long & Murry, 1995). Butler et al. (2014b) revisited the taxonomy of the Bavarian phytosaurs and concluded that “Paleorhinus” angustifrons was a valid species. Kammerer et al. (2015) provided evidence that “Paleorhinus” is a subjective junior synonym of Parasuchus, resulting in the new combination Parasuchus angustifrons. The skull of Parasuchus angustifrons is very well preserved, though somewhat dorsoventrally compressed, and exposes informative features of the palate that are not usually visible or preserved in other basal phytosaurs (BSPG 1931 X 502).

Nicrosaurus kapffi (Meyer, 1860)

Age. Norian, Late Triassic (Hungerbühler, 1998).

Localities. Probably from Heslacher Wand, Stuttgart-Heslach, Baden-Württemberg, Germany (type locality); central Wüttemberg Heslach, Kaltental, Gaisburg, Backnang, and (probably) Degerloch villages on the periphery of Stuttgart and Sindelfingen, Baden-Württemberg, Germany (Hungerbühler, 1998).

Stratigraphic horizon. Löwenstein Formation (Hungerbühler, 1998).

Lectotype. SMNS 4060 and SMNS uncat. no. 15: snout fragment and anterior half of lower jaw (Hungerbühler, 1998).

Paralectotype. SMNS 54708: anterior half of left premaxilla (Hungerbühler, 1998).

Referred material. Multiple specimens described and/or listed by Huene (1923) and Hungerbühler (1998), and housed in the palaeontological collections of the SMNS, NHMUK PV and GPIT.

Diagnosis. Hungerbühler (1998: 41) distinguished Nicrosaurus kapffi from other phytosaurs by the following combination of features: continuous prenarial crest from the external naris to the end of the downturned tip of the snout; and prenarial crest straight or convex at about the level of the skull roof.

Remarks. Nicrosaurus kapffi has a long, complicated taxonomic history that has been summarized by Hungerbühler (1998). The species has been described by Huene (1923), and its cranial anatomy was described in detail by Hungerbühler (1998) and Hungerbühler (2000).

Smilosuchus spp.

Age. Early–middle Norian, Late Triassic (Irmis et al., 2011).

Occurrence. Chinle and Tecovas formations, Arizona and Texas, USA (Long & Murry, 1995).

Species. Smilosuchus gregorii (Camp, 1930), Smilosuchus adamanensis (Camp, 1930), and Smilosuchus lithodendrorum (Camp, 1930) (sensu Stocker, 2010).

Material. Several skulls and postcrania listed by Long & Murry (1995: 227, 299) and housed in the palaeontological collections of the UCMP, USNM, AMNH, UMMP and PPHM.

Synapomorphies. Stocker (2010) found the following synapomorphies for the genus Smilosuchus: ventral margin of squamosal gently sloping anteroventrally from posterior edge of posterior process to opisthotic process; and squamosal fossa extends to posterior edge of squamosal.

Remarks. The species that compose the genus Smilosuchus represent some of the best known phytosaurs and have been long used as representatives of the anatomy of the group, for example regarding hindlimb anatomy (e.g., Parrish, 1986; Sereno & Arcucci, 1990; Sereno, 1991). Smilosuchus gregorii has been used as a species-level representative of phytosaur morphology in previous phylogenetic analyses (e.g., Nesbitt, 2011). The genus has been scored here as a single terminal because the three different species are anatomically conservative and together they provide a fairly complete anatomical record of the skeleton.

Ornithosuchus longidens (Huxley, 1877)

Age. Late Carnian–earliest Norian, early Late Triassic (Rogers et al., 1993; Langer, 2005; Furin et al., 2006; Martínez et al., 2011).

Localities. West (type and referred locality), Spynie, East and Findrassie quarries, Elgin, Scotland, UK (Walker, 1964).

Stratigraphic horizon. Lossiemouth Sandstone Formation (Walker, 1964).

Holotype. EM 1R: large right maxilla.

Referred material. 19–20 specimens listed by Walker (1964: 55–57) and housed in the palaeontological collections of the EM, GSM, MANCH and NHMUK PV.

Diagnosis. Sereno (1991: 12) listed the following autapomorphies that diagnose Ornithosuchus longidens: minor cranial ornamentation; maxilla with free posterior prong; postorbital with strong central horizontal crest; ventral margin of posterior lower jaw concave and elevated; and surangular foramen positioned near surangular-angular suture.

Remarks. See comments in Nesbitt (2011: 22).

Riojasuchus tenuiscepsBonaparte, 1967

Age. Middle Norian, middle Late Triassic (Kent et al., 2014).

Locality. Quebarada de los Jachalleros, La Rioja, Argentina (Bonaparte, 1967; Bonaparte, 1972; Baczko & Desojo, 2016).

Stratigraphic horizon. Upper levels of the Los Colorados Formation, Ischigualasto-Villa Unión Basin (Bonaparte, 1967; Bonaparte, 1972).

Holotype. PVL 3827: complete skull, cervical, dorsal, sacral and caudal vertebrae, scapula, coracoid, humerus, distal portion of the radius and ulna, partial manus, ilium, pubis, femur, tibia, fibula, and nearly complete pes.

Referred material. PVL 3828: nearly complete skull, cervical, dorsal, sacral and caudal vertebrae, scapula, coracoid, humerus, ulna, radius, pubis, ischium, ilium, femur, tibia, fibula, and calcaneum; PVL 2826: cervical, dorsal, sacral and caudal vertebrae, coracoids, scapula fragments, humerus, ulna, radius, ilium, femur, and tibia; and PVL 3814: vertebrae, humerus, and tibia.

Diagnosis. Baczko & Desojo (2016) listed the following autapomorphies that diagnose Riojasuchus tenuisceps: deep antorbital fossa with the anterior and ventral edges almost coinciding with the same edges of the maxilla itself; suborbital fenestra equal in size to the palatine-pterygoid fenestra; and atlantal neural arch bases contact at the midline. These authors distinguished Riojasuchus tenuisceps from all other archosauriforms by the following combination of features: strongly downturned premaxilla; three premaxillary teeth; seven maxillary teeth; second and third teeth on dentary hypertrophied; two-tooth diastema between premaxilla and maxilla; nasal-prefrontal contact absent; jugal with vertical process separating antorbital fenestra from infratemporal fenestra; orbit with ventral point surrounded by V-shaped dorsal processes of the jugal; posterolateral process of the parietals anteriorly inclined greater than 45°; reduced supratemporal fenestra; L-shaped infratemporal fenestra; presence of a palatine-pterygoid fenestra; lower jaws shorter than skull; presence of a first small tooth anterior to the two hypertrophied teeth; anterior end of the dentary dorsally expanded; dentary-splenial symphysis present along one-third of the lower jaw; sharp surangular shelf; presence of a surangular foramen; ventral keel of cervical vertebrae extends ventral to the centrum rims; pubis longer than 70% of femoral length; anterior trochanter (=M. iliofemoralis cranialis insertion) forms a steep margin with the shaft but is completely connected to the shaft; ventral astragalocalcanear articular surface concavoconvex with concavity on astragalus; and metatarsal V without “hooked” proximal end.

It should be noted that the last character-state (metatarsal V without hooked proximal end) occurs in Ornithosuchus longidens (Walker, 1964) but not in Riojasuchus tenuisceps, which possesses a distinctly medially hooked proximal end in the metatarsal V (PVL 3827).

Remarks. See comments in Nesbitt (2011: 22). Baczko & Desojo (2016) recently described in detail the craniodental anatomy, including a digitally reconstructed endocast of the braincase, of Riojasuchus tenuisceps.

Nundasuchus songeaensis Nesbitt et al., 2014

Age. Late Anisian, early Middle Triassic (Nesbitt et al., 2010).

Locality. Z41 locality, between the Ndatira and Njalila rivers, southwestern Tanzania (Nesbitt et al., 2014).

Stratigraphic horizon. Lifua Member, Manda beds, Ruhuhu Basin (Nesbitt et al., 2014).

Holotype. NMT RB48: mostly disarticulated skeleton, including partial right pterygoid, nearly complete right dentary, right splenial, right surangular, isolated teeth, atlas intercentrum, two articulated middle cervical vertebrae, two articulated middle dorsal vertebrae, posterior-most dorsal vertebrae, sacrum and sacral ribs, anterior-most caudal vertebrae, dorsal ribs, gastralia, articulated and isolated paramedian osteoderms, interclavicle, parts of both clavicles, complete left and right scapulae, right coracoid, left humerus, both pubes, both femora, left fibula, proximal and distal portions of left tibia, left astragalus, left calcaneum, left fourth tarsal, left metatarsals (except fourth), distal ends of right metatarsals III, IV, and V, numerous isolated phalanges, partial ungual, and many bone fragments (Nesbitt et al., 2014).

Diagnosis. Nesbitt et al. (2014: 1358) diagnosed Nundasuchus songeaensis on the basis of the following combination of character-states (autapomorphies indicated with an asterisk): multiple anteroposteriorly oriented rows of pterygoid teeth on the palatal process of the pterygoid; anterodorsally oriented grooves on the medial side of the dentary just ventral to dentition*; distinct, lateral expansion of the surangular ridge into a dorsally opening shelf*; paramedian ridges (two total) on the ventral side of the posterior presacral (dorsal) vertebrae*; hyposphene-hypantrum intervertebral articulations in the presacral vertebrae; lateral expansions at the distal margins of the presacral neural spines; two sacral vertebrae; five or six small knobs aligned in an anteroventral row on the lateral side of the coracoid*; short pubic apron; astragalar facet of the calcaneum continuous with a hemicylindrical fibular facet; and anteriorly tapering osteoderms that are arranged in paramedian rows dorsal to the neural spines.

Remarks. Nundasuchus songeaensis was recently named by Nesbitt et al. (2014) and recovered as one of the earliest branching suchians, to the exclusion of gracilisuchids, Revueltosaurus and aetosaurs. Nundasuchus songeaensis possesses an interesting combination of features, with a number of plesiomorphic archosaur character-states that optimize as autapomorphies (Nesbitt et al., 2014).

Turfanosuchus dabanensis Young, 1973c

Age. Anisian, early Middle Triassic (Fröbisch, 2009).

Locality. Taoshuyuanzi, about 30 km northwest of Turfan Basin, Xinjiang Autonomous Region, China (Wu & Russell, 2001).

Stratigraphic horizon. Vertebrate Fossil Bed IV, lower Karamayi Formation (=Kelamayi Formation) (Wu & Russell, 2001).

Holotype. IVPP V3237: partial skeleton (currently mounted in plaster, with the exception of the skull, and several bones reconstructed with plaster).

Emended diagnosis. Turfanosuchus dabanensis is a gracilisuchid archosaur that differs from other archosauromorphs in the following combination of character-states (autapomorphies indicated with an asterisk): nasal with a narrow anterolateral process that bifurcates to receive the postnarial process of premaxilla; maxilla possesses a tall, triangular dorsal process with clear dorsal apex formed by discrete expansion of the posterior end of the horizontal process and broadly contributes to the posteroventral border of the antorbital fenestra in lateral view*; upper temporal bar with a longitudinal, thick ridge on the dorsolateral surface; squamosal with a tapering anteroventral process that subdivides the infratemporal fenestra*; dentary with elongate posteroventral process longer than posterodorsal process; and posterolateral surface of surangular highly concave; and angular excluded by surangular-dentary contact from margin of external mandibular fenestra.

Remarks. Turfanosuchus dabanensis was erected by Young (1973c) and redescribed by Wu & Russell (2001). Nesbitt (2011: 21, 22) discussed the history of the alternative phylogenetic positions that the species had occupied in different phylogenetic analyses and its importance in early archosaur evolution. Butler et al. (2014a) recently recovered Turfanosuchus dabanensis as more closely related to Gracilisuchus stipanicicorum and Yonghesuchus sangbiensis, within the new family Gracilisuchidae.

Gracilisuchus stipanicicorum Romer, 1972c

Age. Early Carnian, early Late Triassic (Marsicano et al., 2015).

Locality. Two kilometres north of the Río Chañares, Los Chañares type locality, near the mouth of Chañares River, Talampaya National Park, La Rioja Province, Argentina (Romer, 1972c).

Stratigraphic horizon. Lower member of the Chañares Formation (sensu Fiorelli et al., 2013), Ischigualasto-Villa Unión Basin (Romer, 1972c).

Holotype. PULR 08: partial skeleton, including fairly complete skull and lower jaw.

Referred material. CRILAR-Pv 490: partial postcranium; MCZ 4116A (in part): partial skull, an incomplete articulated caudal vertebral series, and articulated ischia; MCZ 4117: almost complete and well preserved skull; MCZ 4118, partial skull, articulated cervical series (from axis to cervical 6) and osteoderms, a series of three articulated cervico-dorsal vertebrae, a series of six cervico-dorsal vertebrae articulated with ribs, a dorsal series of at least nine elements; PVL 4597: partial skeleton, including nearly complete skull and lower jaw; PVL 4612, nearly complete skull articulated with the left hemimandible (modified from Lecuona & Desojo, 2011).

Diagnosis. Lecuona & Desojo (2011: 106) diagnosed Gracilisuchus stipanicicorum by the following unique combination of character-states (autapomorphies indicated with an asterisk): large antorbital fenestra occupying approximately 0.3 of the anteroposterior length of the skull table (measured from the anterior end of the premaxilla to the posterior end of the parietals); large antorbital fossa occupying 0.4 of the length of the skull table; presence of a postfrontal and a small postparietal; anterior ramus of squamosal laterally extended; interparietal suture partially obliterated; narrow occipital portion of the parietals; postzygapophyseal facet of the axis horizontal, posteriorly directed, and facing ventrally*; high and vertical anterior border of the axial neural spine*; presence of a ventral longitudinal median keel on axial centrum; poor development of ventral keel on the cervical vertebrae; circular depression on the mid-dorsal region of the neural arch of cervical vertebrae; spine table in posterior cervical vertebrae; lack of a well-defined acetabular surface on the pubis; L-shaped lamina on proximal pubic apron; ischiadic symphysis proximally located*; femur longer than tibia; knob-shaped iliofibular trochanter; and two paramedian osteoderms per vertebra.

Remarks. Romer (1972c) erected and briefly described Gracilisuchus stipanicicorum based on several skeletons collected from the Chañares Formation of northwestern Argentina. The phylogenetic history of the species has been summarized by Nesbitt (2011: 19). Butler et al. (2014a) recovered Gracilisuchus stipanicicorum as a member of the new family Gracilisuchidae, together with the Chinese species Turfanosuchus dabanensis and Yonghesuchus sangbiensis. The pelvic girdle and hindlimb anatomy of the species was redescribed in detail by Lecuona & Desojo (2011).

Aetosauroides scagliai Casamiquela, 1960

Age. Late Carnian–earliest Norian, early Late Triassic (Rogers et al., 1993; Furin et al., 2006; Martínez et al., 2011).

Localities. Hoyada de Ischigualasto, Ischigualasto Provincial Park, San Juan Province, Argentina (type and referred localities); and localities in the Rio Grande do Sul State, Brazil (Casamiquela, 1960; Casamiquela, 1961; Casamiquela, 1967; Desojo & Ezcurra, 2011). The exact stratigraphic provenance of the historical Argentinean specimens described by Casamiquela cannot be determined because of the lack of precise occurrence information. Nevertheless, Martínez et al. (2012) provided a census of a total of 20 specimens of Aetosauroides and determined that they come from the Cancha de Bochas and the lower Valle de la Luna members of the Ischigualasto Formation.

Stratigraphic horizons. Lower levels of the Ischigualasto Formation, Agua de la Peña Group, Ischigualasto-Villa Unión Basin, Argentina, and Santa Maria Sequence 2, Hyperodapedon AZ, Rosário do Sul Group, Paraná Basin, Brazil (Casamiquela, 1960; Casamiquela, 1961; Casamiquela, 1967; Desojo & Ezcurra, 2011).

Holotype. PVL 2073: partial, articulated postcranial skeleton.

Referred material. PVL 2052: natural external mould of the right anterior half of the skull and a portion of the right hemimandible, proximal half of both femora, both tibia, distal end of right fibula, both articulated tarsals and pes, articulated dorsal and ventral carapaces, and some isolated paramedian osteoderms; PVL 2059: partial skeleton, including an incomplete skull and lower jaw; PVL 2091 (holotype of Argentinosuchus bonapartei Casamiquela, 1961): atlas, axis, and third to fifth cervical vertebrae in articulation, with some cervical ribs and osteoderms, a right paramedian osteoderm, two right lateral osteoderms, three ventral osteoderms, three appendicular osteoderms, left humerus, proximal end of left radius and ulna, and seven fragments of osteoderms; PVSJ 326: skull without lower jaw, two posterior or mid-dorsal vertebrae, eight anterior and mid-caudal vertebrae, right humerus, ulna, and radius, right femur, tibia, fragmentary fibula, proximal tarsals, metatarsals I–IV, and three non-ungual phalanges, and one paramedian and one appendicular osteoderm; MCP 13a-b-PV (holotype of “Aetosauroides subsulcatus” Zacarias, 1982): articulated partial dorsal and ventral carapaces, several isolated paramedian and ventral osteoderms, six complete and articulated middle and posterior dorsal vertebrae, some fragments of dorsal vertebrae and ribs; UFSM 11070a (a single specimen that was collected and housed by three different institutions: MCP, UFRGS, and UFSM): partial braincase, distal half of metatarsal III or IV, one anterior cervical centrum, several dorsal and caudal vertebrae, including a series of four articulated dorsals with associated paramedian osteoderms and dorsal ribs, right calcaneum, several fragments of dorsal ribs, a chevron, and several paramedian osteoderms (housed at MCP), fragmentary two dorsal vertebrae, complete right articulated metacarpus, three non-ungual phalanges, right femur, probable distal end of tibia, dorsal ribs, left calcaneum, several paramedian and appendicular osteoderms (housed at UFRGS), seven dorsal and caudal vertebrae, dorsal ribs, and several paramedian osteoderms associated with rhynchosaur remains (housed at UFSM) (modified from Desojo & Ezcurra, 2011).

Diagnosis. Desojo & Ezcurra (2011: 598) distinguished Aetosauroides scagliai from other known pseudosuchian archosaurs by the following unique combination of apomorphies (autapomorphy indicated with an asterisk): maxilla excluded from the margin of the external naris; ventral margin of dentary convex and without sharp inflexion; tooth crowns with straight distal margin and without constriction between root and crown, denticles, and wear facets; cervical and dorsal centra with oval fossae ventral to the neurocentral suture on the lateral sides of the centra; middle and posterior dorsals with well-developed posterior centrodiapophyseal lamina; and middle and posterior dorsals posterolaterally divergent postzygapophyses, and ratio between the entire length of the postzygapophyses and the width between the distal-most tips of the postzygapophyses equal or lower than 0.75*.

Remarks. Aetosauroides scagliai was originally erected by Casamiquela (1960) and described by the same author in a series of papers (Casamiquela, 1960; Casamiquela, 1961; Casamiquela, 1967). Heckert & Lucas (2000) proposed that Aetosauroides was a subjective junior synonym of Stagonolepis, but this hypothesis was subsequently rejected by Desojo (2005) and Desojo & Ezcurra (2011). Desojo & Ezcurra (2011) also proposed that Aetosauroides “subsulcatus” from Brazil was a subjective junior synonym of Aetosauroides scagliai and extended the geographic distribution of this species to both countries. Aetosauroides scagliai has been recently recovered as the basalmost aetosaur (Desojo, Ezcurra & Kischlat, 2012; Roberto-Da-Silva et al., 2014; Heckert et al., 2015; Parker, 2016). As a result, the combination of well-known anatomy and its basal position within the group makes Aetosauroides scagliai a key representative of aetosaur morphology in phylogenetic analyses.

Batrachotomus kupferzellensis Gower, 1999

Age. Late Ladinian, late Middle Triassic (Gower, 1999; Schoch, 2002).

Locality. Kupferzell, Hohenlohe region of northern Baden-Württemberg (type and referred localities); and Vellberg-Eschenau and Crailsheim, about 10 and 30 km east of Schwäbisch Hall, Baden-Württemberg, Germany (Gower, 1999; Gower & Schoch, 2009).

Stratigraphic horizon. Lower Keuper, Erfurt Formation (Gower, 1999; Gower & Schoch, 2009).

Holotype. SMNS 52970: premaxillae, maxillae, nasals, frontal, postfrontals, parietals, squamosals, postorbitals, jugals, quadrates, dentaries, surangulars, articulars, right lacrimal, right prefrontal, left quadratojugal, left ectopterygoid, left prearticular, isolated teeth, three dorsal, one sacral and three caudal vertebrae, one dorsal osteoderm, right ilium, and a left femur, tibia and fibula (Nesbitt, 2011). A proximal fragmentary right femur catalogued under this number is slightly smaller than the complete right element and questionably belongs to the same individual as the bulk of the holotypic material (Gower & Schoch, 2009).

Referred material. Multiple specimens listed by Gower & Schoch (2009: Appendix I) and housed in the collections of the SMNS and MHI.

Diagnosis. Gower & Schoch (2009: 103, 104) used the following unique combination of features to diagnose Batrachotomus kupferzellensis: short and robust postnasal process of the premaxilla; maxillary palatal process well dorsal to the ventral edge of the bone; kinked anterodorsal maxillary border that contributes to the border of the naris; short ascending process of the maxilla; short and broad dorsal part of the prefrontal; concave anterodorsal edge of the axial neural spine; dorsal vertebrae lacking greatly elongated neural spines; three sacral vertebrae with relatively long, strongly ventrally deflected sacral ribs; and ilium with a subvertical instead of anterodorsally trending (onto anterior iliac process) rugose iliac ridge above the acetabulum, lacking a waisted ilium (sensu Nesbitt, 2005), and lacking coossification among sacral vertebrae.

Remarks. See comments in Nesbitt (2011: 36).

Prestosuchus chiniquensis Huene, 1938

Age. Ladinian–early Carnian, late Middle–early Late Triassic (Langer et al., 2007; Marsicano et al., 2015).

Localities. Weg Sanga (type locality), and Sanga Pascual of the Pinheiros locality, “Posto da Gasolina”, and Sanga da Árvore (Baum Sanga) of the Xiniquá region, Rio Grande do Sul State, Brazil (Huene, 1938; Langer et al., 2007; Mastrantonio et al., 2013).

Stratigraphic horizon. Santa Maria 1 Sequence, Dinodontosaurus AZ (Huene, 1938; Langer et al., 2007; Mastrantonio et al., 2013).

Lectotype. BSPG 1933L 1-3/5-11/28-41/41: splenial, anterior portion of the surangular, anterior portion of the angular, prearticular, right partial maxilla, fragmentary dentary, three incomplete cervical vertebrae, fragmentary ribs, one sacral vertebra, two sacral ribs, five anterior caudal vertebrae with chevron bones, 14 middle and posterior caudal vertebrae, left and right scapulocoracoid, interclavicle and clavicle, distal end of left humerus, proximal and distal ends of right humerus, distal end of radius, fragmentary ulna, one manual phalanx, incomplete ilium, fragmentary ischia, pubes, and complete left hind limb.

Paralectotype. BSPG 1933L/7: articulated vertebral sequence composed of two sacral vertebrae with sacral ribs, incomplete last dorsal and first caudal vertebrae, dorsal portion of the right ilium, and a series of osteoderms articulated with the neural spines.

Referred material. UFRGS 0156-T: complete skull, much of the presacral axial column and articulated osteoderms; UFRGS 0152-T: maxillae, nasals, quadrate, partial quadratojugal, complete braincase, parietal, ectopterygoid, partial pterygoid, jugal, squamosal, anterior portion of the dentary, prearticular, articular, cervical, dorsal, sacral, and caudal vertebrae, osteoderms, scapula, coracoid, humerus, proximal portion of the ulna, complete pelvic girdle, femora, tibia, fibula, calcaneum, pes, chevrons (Nesbitt, 2011); UFRGS-PV-0629-T: partial skeleton (Mastrantonio et al., 2013); MCP-146: a complete pelvic girdle with the last dorsal, two sacral, and three caudal vertebrae preserved in articulation (Bonaparte, 1984; Lacerda et al., 2016); MCZ 4167: partial skeleton (Lacerda et al., 2016); and CPEZ-239b: partial skeleton of a probable juvenile individual (associated with the holotype of Archeopelta arborensis) (Lacerda et al., 2016).

Diagnosis. Two autapomorphies were listed by Desojo & Rauhut (2008): anterior notch between the scapula and coracoid; and a longitudinal ridge on the dorsal surface of the ilium. Nevertheless, a revision of the taxonomy of the Brazilian rauisuchians is currently in preparation (JB Desojo, pers. comm., 2015) and it is necessary to wait for such information to propose an emended diagnosis for the genus and species (Lacerda et al., 2016).

Remarks. See comments in Nesbitt (2011: 33)—for Prestosuchus chiniquensis, UFRGS 0156-T and UFRGS 0152-T—and Lacerda et al. (2016).

Dimorphodon macronyx (Buckland, 1829)

Age. Hettangian–Sinemurian, Early Jurassic (Hallam, 1960).

Occurrence. Lower Lias, Lyme Regis, Dorset, England.

Holotype. NHMUK PV R1034: partial postcranium.

Referred material. NHMUK PV R1035: partial skull and fragmentary postcranium; NHMUK PV R41212: partial skeleton; and several specimens listed by Padian (1983: Table 1) and housed in the collection of the YPM.

Emended diagnosis. Dimorphodon macronyx is a pterosaur distinguished from other archosauromorphs by the following combination of features (autapomorphies indicated with an asterisk): strongly heterodont dentition, with long, slender, trenchant and sharp-pointed anterior teeth and small, closely packed and lancet-shaped posterior teeth*; skull without cresting on the skull roof; external naris that occupies most of the lateral surface of the snout; dentary without a downturned anterior end; tall and slender anterodorsal process of the jugal, forming approximately half of the posterior border of the antorbital fenestra*; and lower jaw with an external mandibular fenestra.

Remarks. See comments in Nesbitt & Hone (2010) and Nesbitt (2011: 43).

Lagerpeton chanarensis Romer, 1971b

Age. Early Carnian, early Late Triassic (Marsicano et al., 2015).

Localities. Approximately 4 kilometres east of the Los Chañares locality, exposures on the northern flank of the north branch of the Chañares River (type locality); and Los Chañares locality, Talampaya National Park, La Rioja Province, Argentina (Sereno & Arcucci, 1994a).

Stratigraphic horizon. Probably lower member of the Chañares Formation (sensu Fiorelli et al., 2013), Ischigualasto-Villa Unión Basin.

Holotype. PULR 06: articulated right hindlimb.

Referred material. PVL 4619: articulated sacrum, pelvis, and partial right and left hindlimbs; PVL 4625: articulated vertebral column including dorsal, sacral, and anterior caudal vertebrae, left pelvis, and left femur; and MCZ 4121: partial right and left femora. Nesbitt (2011) listed a proximal left femur (PVL 5000) among the referred specimens of Lagerpeton chanarensis, but this catalogue number corresponds to a notoungulate mammal (J Powell, pers. comm., 2015).

Diagnosis. Sereno & Arcucci (1994a: 386) diagnosed Lagerpeton chanarensis as a slender-limbed ornithodiran archosaur characterized by: posterior dorsal vertebrae with anterodorsally inclined neural spines; first sacral vertebra with fan-shaped rib extending anterodorsally to the tip of the preacetabular process of the ilium; iliac blade with sinuous dorsal margin; preacetabular process laterally convex with anterior end directed anteromedially; ischial peduncle of ilium recessed; band-shaped eminence passing posterodorsally across lateral surface of postacetabular process; ischium with broad convex ventromedial flange and vertically deep puboischial suture; distal ischial blades horizontal; proximal end of pubis with subtriangular lateral fossa; pubic shaft deflected medially distal to ambiens process; proximal end of femur with flat anteromedial surface; deep femoral head with hook-shaped medial extension; elongate aliform fourth trochanter; distal end of femur with large fibular condyle; astragalus with tongue-shaped posterior ascending process; vascular depression at base of anterior ascending process of astragalus absent; ascending process of the astragalus broadly overlapping tibia and fibula; astragalus and calcaneum co-ossified; calcaneum with flat distal surface canted anteroproximally relative to astragalar distal surface; pedal digit I short relative to digits II–IV, with metatarsal I shorter than all other metatarsals and shorter than phalanx I-1; metatarsal I with laterodistally bevelled proximal end articulating medially with metatarsal II; pedal digit II short relative to digits III and IV with metatarsal II less than two-thirds metatarsal III and only half as long as metatarsal IV; and pedal digit IV and metatarsal IV longer than pedal digit III and metatarsal III, respectively.

Remarks. See comments in Nesbitt (2011: 44).

Marasuchus lilloensis (Romer, 1971b)

Age. Early Carnian, early Late Triassic (Marsicano et al., 2015).

Locality. Southwest part of the Los Chañares locality, Talampaya National Park, La Rioja Province, Argentina (Bonaparte, 1975; Sereno & Arcucci, 1994b).

Stratigraphic horizon. Lower member of the Chañares Formation (sensu Fiorelli et al., 2013), Ischigualasto-Villa Unión Basin.

Holotype. PVL 3871: partial articulated skeleton including the posterior portion of the vertebral column (from the last dorsal vertebra to the 25th caudal vertebra), left scapulocoracoid, humerus, radius, ulna, fragmentary right pelvis, left ilium, left pubis, and partial right and left hindlimbs.

Referred material. PVL 3870: partial skeleton including the maxilla and partial braincase, vertebral column from the atlas to the anterior caudal vertebrae, articulated pelvis and hindlimbs lacking only the distal phalanges and unguals; PVL 3872: partial braincase and articulated vertebral column from the atlas to the ninth presacral vertebra; PVL 4670: articulated anterior caudal vertebrae with chevrons; PVL 4671: articulated anterior caudal vertebrae with chevrons; and PVL 4672: articulated vertebral column from atlas to the 17th presacral vertebra.

Diagnosis. Sereno & Arcucci (1994b: 56) diagnosed Marasuchus lilloensis as a dinosauriform archosaur characterized by: anterodorsally projecting posterior cervical neural spines (presacral vertebrae 6–9); marked fossa ventral to the transverse processes in the posterior cervicals and anterior dorsal vertebrae (presacral vertebrae 6 through 10 or 12); subtriangular neural spines in middle and posterior dorsal vertebrae that contact each other anteriorly and posteriorly; middle caudal centra twice the length of anterior caudal centra; elongate anterior chevrons that are more than three times the length of the first caudal centrum; broad scapular blade; transversely concave distal pubic blade; and transversely narrow fibular articular surface on calcaneum.

Remarks. See comments in Nesbitt (2011: 45, 46).

Lewisuchus admixtus Romer, 1972d

Age. Early Carnian, early Late Triassic (Marsicano et al., 2015).

Locality. Los Chañares locality, Talampaya National Park, La Rioja Province, Argentina (Romer, 1972d).

Stratigraphic horizon. Lower member of the Chañares Formation (sensu Fiorelli et al., 2013), Ischigualasto-Villa Unión Basin.

Holotype. PULR 01: incomplete left and right maxillae with teeth; posterior portion of the skull, including the left laterotemporal region: jugal, postorbital, quadratojugal, squamosal, and quadrate; braincase, comprising supraoccipital, basioccipital, otoccipital, laterosphenoid, parabasisphenoid and prootic; articulated left pterygoid and ectopterygoid; cervical vertebrae from atlas to the seventh vertebra, 11 dorsal vertebrae, nine proximal to middle caudal vertebrae; both scapulocoracoids and humeri; and incomplete tibiae. Bones previously referred to the holotype were later reassigned to indeterminate proterochampsids by Bittencourt et al. (2014), including an isolated dentary and pedal elements. Although this dentary does not match the morphology of a proterochampsid dentary because it is dorsally curved along its entire length, the composition of the hypodigm of Lewisuchus admixtus proposed by Bittencourt et al. (2014) is followed here until more evidence is available.

Diagnosis. Bittencourt et al. (2014: 191) diagnosed Lewisuchus admixtus as a dinosauromorph that can be distinguished by the following combination of features (autapomorphies indicated with an asterisk): extremely elongated skull; supraoccipital nearly horizontal; three foramina posterior to the metotic strut*; anteroposteriorly extending rugose ridge on the middle height of the lateral surface of the axial neural spine*; postzygodiapophyseal lamina of the posterior cervical vertebrae projecting posteriorly to the tip of the postzygapophysis; and the presence of a single row of scutes associated to the distal tip of the cervical and dorsal neural spines.

Remarks. The phylogenetic history of Lewisuchus admixtus has been summarized by Nesbitt (2011: 46). Bittencourt et al. (2014) redescribed in detail the holotype and single known specimen of the species. A recently collected new dinosauriform specimen from the Chañares Formation seems to indicate that Lewisuchus admixtus is a subjective senior synonym of Pseudolagosuchus major (Novas, Agnolín & Ezcurra, 2015), as suggested by Arcucci (1997), Arcucci (1998) and Nesbitt & Hone (2010). Lewisuchus admixtus is scored here solely based on the holotype specimen PULR 01 until a detailed discussion of its synonym with Pseudolagosuchus major based on strong evidence is published.

Silesaurus opolensis Dzik, 2003

Age. Late Carnian, early Late Triassic (Dzik, 2001).

Occurrence. Clay pit in Krasiejów, Opole, Silesia, Poland (Dzik, 2003).

Holotype. ZPAL AbIII/361: dentaries, braincase, pterygoid, frontals, quadrate, surangular, nearly complete presacral column, sacrum, caudal vertebrae, scapulocoracoid, radii, ulnae, complete pelvic girdle and hindlimbs.

Referred material. Multiple specimens listed by Dzik (2003), Dzik & Sulej (2007), and Piechowski & Dzik (2010) and housed in the collection of the ZPAL.

Diagnosis. Dzik (2003: 560) diagnosed Silesaurus opolensis on the basis of the following combination of features: beak on the dentaries; small number of teeth (11–12 in both dentary and maxilla); 25 presacral vertebrae with 9–11 cervicals; elongate, gracile front limbs; and digit I of pes reduced to vestigial metatarsal that probably lacked phalanges. Dzik (2003) originally interpreted that the sacrum of Silesaurus opolensis was composed of four fused vertebrae, but subsequently Dzik & Sulej (2007) reinterpreted the species as having three sacral vertebrae.

Remarks. See comments in Nesbitt (2011: 49).

Heterodontosaurus tucki Crompton & Charig, 1962

Age. Hettangian–Toarcian, Early Jurassic (Norman et al., 2011).

Localities. Behind Tyindini trading store (type locality), and northern slope of Kromspruit Mountain and site 18a of the Kromspruit 9 Farm, Herschel Distrinct, Eastern Cape Province, South Africa (Norman et al., 2011).

Stratigraphic horizons. Clarens Formation and upper part of the Elliot Formation, Stormberg Series (Norman et al., 2011).

Holotype. SAM-K-337: partial skull and lower jaw.

Referred material. SAM-PK-K1332: fairly complete, articulated skeleton; SAM-PK- K1334: partial left maxilla with associated fragments of jugal and lacrimal; and SAM-PK-K10487: partial skull and lower jaw (Norman et al., 2011).

Diagnosis. Norman et al. (2011: 187) listed the following character-states to diagnose Heterodontosaurus tucki (autapomorphies indicated with an asterisk): deep buccal emargination formed by a strongly dorsoventrally compressed and transversely expanded maxillary ridge, which forms the ventral margin of the external antorbital fenestra and is thickened along its lateral margin*; antorbital fossa extends posteriorly to form a channel on the external surface of the jugal*; quadratojugal forms a thin wing that overlaps the entire external surface of the quadrate (contacting the squamosal dorsally and terminating ventrally just above the articular condyle) and contacts the jugal via a narrow bridge of bone*; quadratojugal has a constricted scarf suture with the jugal*; narrow and obliquely orientated ventral jugal projection closely aligned against the lateral surface of the lower jaw*; prominent laterally expanded ‘boss’ on the jugal*; sharply defined curved ridge on the external surface of the postorbital that is continuous with a similar ledge on the dorsolateral margin of the squamosal*; remnants of intracranial pneumatism preserved as pits on the paroccipital process and quadrate, and as sinuses on the jugal boss and anteromedial process of the maxilla*; narrow and deep pterygoid flanges lie close to the medial surface of the lower jaw (forming a slot-like guide with the ventral process of the jugal)*; paroccipital wings perforated by a discrete vascular/neural canal*; basisphenoid flanges are large, oblique and extend medial to the pterygoids and enclose narrow fossae on either side of the ventral midline of the braincase; surangular develops two finger-like rami that form much of the dorsal margin of the coronoid eminence anterior to the jaw joint*; elongate, slot-shaped surangular foramen*; broad depression on the lateral surface of the angular*; premaxillary and dentary caniniforms have fine, blunt, serrations (six per mm) running down their posterior margins; premaxillary caniniform lacks serrations along its anterior edge; dentary caniniform has widely spaced, rounded denticulations running down the upper portion of its anterior edge*; columnar maxillary and dentary teeth have crowns that are only slightly expanded either anteroposteriorly or transversely above the root (the ‘cingulum’ and ‘neck’ at the crown root junction are completely absent)*; labial surface of maxillary crowns possesses three prominent ridges that separate equal-sized, clearly defined excavated regions*; lingual surface of dentary crowns displays a mesially offset principal ridge and crown margins that create subequal adjacent crown areas*; extensive wear facets on the upper and lower dentitions display a warp because successive teeth are worn at differing angles*; axial vertebral column: 21 vertebrae (9 cervical, 12 dorsal)*, sacrum: 6 fused vertebrae*, caudal vertebrae: 34+; prominent epipophyses present on anterior cervical postzygapophyses*, ossified tendons distributed across the neural spines of dorsal and sacral vertebrae only; scapular blade narrow and elongate with expanded distal (extrascapular) portion; humerus with a large deltopectoral crest and large entepicondyle*; humerus lacks a posterior (olecranon) fossa; ulna with prominent olecranon; manus length more than 40% of the combined length of humerus and radius; nine carpal bones; manus digits I–III parallel, digits IV–V reduced in size and divergent; penultimate phalanges of digits II and III more elongated than the proximal phalanges; extensor pits present on the dorsal surface of distal end metacarpals and phalanges; manual unguals strongly recurved, and with prominent flexor tubercles; ilium with a narrow vertical facet on the ischial peduncle that resembles an avian antitrochanter*; prepubic process short and deep, postpubis as long as ischium; obturator process absent; ischial shaft marked by an elongate lateral ridge that is drawn out to form a prominent lateral shelf along the mid-section of the shaft*; femoral greater and anterior trochanters not separated by a cleft; transverse axis of distal femoral articular surface obliquely orientated; fibula reduced and fused to tibia distally*; astragalus and calcaneum fused*; astragalocalcaneum fused to the distal ends of tibia and fibula*; three distal tarsals present but fused to proximal ends of their metatarsals*; and metatarsals I–IV fused together*.

Remarks. Nesbitt (2011: 50) summarized the phylogenetic history of Heterodontosaurus tucki. Santa Luca (1980) and Galton (2014) described the postcranium of this species, and Norman et al. (2011) described in detail its craniodental anatomy. Sereno (2012) added a substantial new amount of information about the anatomy of Heterodontosarus tucki in the context of an overall revision of heterodontosaurids.

Herrerasaurus ischigualastensis Reig, 1963

Age. Late Carnian-earliest Norian, early Late Triassic (Rogers et al., 1993; Furin et al., 2006; Martínez et al., 2011).

Locality. Hoyada de Ischigualasto, Ischigualasto Provincial Park, San Juan Province, Argentina (Reig, 1963; Novas, 1993; Sereno, 1994; Sereno & Novas, 1994; Martínez et al., 2012).

Stratigraphic horizon. Chancha de Bohas Member, lower levels of the Ischigualasto Formation, Agua de la Peña Group, Ischigualasto-Villa Unión Basin (Reig, 1963; Novas, 1993; Sereno, 1994; Sereno & Novas, 1994; Martínez et al., 2012).

Holotype. PVL 2566: dorsal, sacral, and caudal vertebrae, ilium, pubis, ischium, right femur, metatarsals, phalanges and left astragalus.

Referred material. MACN-Pv 18060 (holotype of “Ischisaurus cattoi” Reig, 1963): partial skeleton, including some cranial bones and elements from all postcranial regions; MCZ 4381: partial pelvic girdle, including both ilia, pubes and proximal end of ischia, and proximal end of left tibia; MCZ 7064: atlas, axis, fragments of at least five dorsal vertebrae, both partial scapulae and coracoids, proximal and distal ends of both humeri, acetabular region of the left ilium, distal halves of both pubes, distal end of the right femur, proximal end of the right tibia and fibula, distal end of the left tibia, and fragments of some pedial phalanges; MLP 61-VIII-2-2: both fairly complete, but poorly preserved pelvic girdle; MLP 61-VIII-2-3: right coracoid, both humeri, fragmentary ulnae, right femur and tibia, and some elements of the pes; PVL 2054: fragments of both pubes and gastralia, most of both femora, tibiae, and fibulae, several metatarsals and phalanges of both pes; PVL 2558a: fragments of both femora, tibiae, and an isolated manual ungual; PVSJ 53 (holotype of “Frenguellisaurus ischigualastensis” Novas, 1986): partial skull and complete lower jaw, axis, fragments of cervical and dorsal vertebrae, 25 mostly articulated distal caudal vertebrae, right scapula and fragmentary coracoid, distal end of right humerus and fragment of left humerus, distal ends of both ulnae, proximal end of left ulna and distal end of radius; PVSJ 104: fragmentary pelvic bones, sacral vertebrae and hindlimb elements; PVSJ 373: well preserved articulated skeleton, lacking skull and most cervical and caudal vertebrae; PVSJ 380: right scapula and nearly complete, articulated right carpus and manus; PVSJ 407: nearly complete articulated skeleton with skull and lower jaw; and PVSJ 461: nearly complete, poorly preserved skeleton.

Diagnosis. Novas (1993: 401) diagnosed Herrerasaurus ischigualastensis on the basis of the following combination of character-states: premaxillary-maxillary fenestra posterior to external naris; anterior end of both antorbital fenestra and antorbital fossa semicircular; ventral border of maxilla sinuous, especially at level of jugal articulation; ridge on lateral surface of jugal; upper part of infratemporal fenestra less than one-third as broad as lower part; deeply incised supratemporal fossa that extends across the medial process of postorbital; ventral process of squamosal subquadrate with lateral depression; quadratojugal overlaps posterodorsal face of quadrate; pterygoid ramus of the quadrate with inturned, trough-shaped ventral margin; dentary with slender posterodorsal process with T-shaped cross-section; surangular with forked anterior process for articulation with posterodorsal process of dentary; apices of neural spines of posterior dorsal vertebrae with pronounced lateral borders; internal tuberosity of humerus projecting proximally and separated from the humeral head by deep groove; humeral entepicondyle ridge-like with anterior and posterior depressions; manus 60% of humerus plus radius length; pubis proximally curved and ventrally oriented; lateral margin of the pubis sinuous in anterior view; ischium with posterior border of postacetabular pedicle forming a right angle with dorsal border of ischial shaft; femur with anteroproximal keel and subcircular muscle scar on anterolateral distal shaft; and tibia shorter than femur (tibia length represents 87–91% of femur length).

Remarks. See comments in Nesbitt et al. (2009: 853) and Martínez et al. (2012: 28).

Character sampling and scorings

The character sampling of the phylogenetic analysis was built by combining the character lists of previous phylogenetic analyses focused on non-archosaurian archosauromorphs and basal archosaurs (e.g., Parrish, 1992; Gower & Sennikov, 1996; Gower & Sennikov, 1997; Dilkes, 1998; Ezcurra, Lecuona & Martinelli, 2010; Ezcurra, Scheyer & Butler, 2014; Nesbitt, 2011; Desojo, Ezcurra & Schultz, 2011; Dilkes & Arcucci, 2012; Trotteyn & Ezcurra, 2014; Ezcurra, Desojo & Rauhut, 2015; Pritchard et al., 2015; Nesbitt et al., 2015) after evaluating the independence between characters (repeated or partially non-independent characters were combined with one another). Furthermore, 96 new characters were added. The complete character list includes 600 characters, comprising 309 cranial (51.5%) and 291 postcranial (48.5%) characters. The following 82 characters represent nested sets of homologies and, as a result, were treated as additive (=ordered): 1, 2, 7, 10, 17, 19–21, 28, 29, 36, 40, 42, 50, 54, 66, 71, 75, 76, 122, 127, 146, 153, 156, 157, 171, 176, 177, 187, 202, 221, 227, 263, 266, 279, 283, 324, 327, 331, 337, 345, 351, 352, 354, 361, 365, 370, 377, 379, 398, 410, 424, 430, 435, 446, 448, 454, 458, 460, 463, 472, 478, 482, 483, 489, 490, 504, 510, 516, 529, 537, 546, 552, 556, 557, 567, 569, 571, 574, 581, 582 and 588. Meristic (i.e., characters with character-states represented by numerical ratios rather than qualitatively described features) or count-based characters were discretized using cluster analyses of the raw values for each sampled individual of each terminal (i.e., ranges were included only if there was intraindividual variation for the ratio) in PAST 2.17c (Hammer, Harper & Ryan, 2001) (see Appendix S1). In the resultant meristic trees, two different states were discretized if the distance between them was larger than the complete internal distance of both clusters. In addition, character-states that were separated from each other by less than 5% of the complete range of the sampled raw ratios were merged together.

Table 1 Sources of scoring for each operational taxonomic unit.

The specimen numbers listed here correspond to specimens studied at first hand.

Terminal	Sources of scorings	
Petrolacosaurus kansensis	Peabody (1952); Reisz (1977) and Reisz (1981)	
Acerosodontosaurus piveteaui	MNHN 1908-32-57; Currie (1980); Bickelmann, Müller & Reisz (2009)	
Youngina capensis	BP/1/2459, 3859; GHG K 106, RS 160; NHMUK PV R5481; SAM-PK-K6205, K7578, K7710, K8565; TM 1490, 3603; Broom (1914); Broom (1921); Broom (1922); Gow (1975); Currie (1981); Evans (1987); Smith & Evans (1996); Dilkes (1998); Gardner, Holliday & O’Keefe (2010)	
Paliguana whitei	AM 3585; Carroll (1975)	
Planocephalosaurus robinsonae	NHMUK PV R9954–R9969, R9971–R9976; Fraser (1982); Fraser & Walkden (1984); Fraser & Shelton (1988)	
Gephyrosaurus bridensis	Evans (1980); Evans (1981); Fraser & Shelton (1988)	
Cteniogenys sp.	Multiple cranial and postcranial bones housed in the NHMUK PV; Evans (1990); Evans (1991)	
Simoedosaurus lemoinei	MNHN.F.BL9022, BL9425, BL9525, BL9626, BR728, BR1009, BR1013, BR1018, BR1348, BR2075, BR1935, BR10199, BR11715, BR12090, BR12091, BR12108, BR12153, BR12154, BR12179, BR12208, R1381, R1413, R2241, R3313, R3404, R3945, R4014; SMNS 58674, 58594, 58566, 58542, 58564, 58540, 58541, 58577, 58593; Sigogneau-Russell & Russell (1978); Sigogneau-Russell (1981)	
Aenigmastropheus parringtoni	UMZC T836; Parrington (1956); Ezcurra, Scheyer & Butler (2014)	
Protorosaurus speneri	BSPG 1995 I 5 (cast of WMsN P47361), 1997 I 12 (cast), 1997 I 13 (cast), AS VII 1207; NHMW 1943I4; USNM 442453 (cast of NMK S 180); SMNS 55387 (cast of Simon/Bartholomäus specimen), 59345 (cast); ZMR MB R2171–R2173 (casts of specimens destroyed probably during WWII); Gottmann-Quesada & Sander (2009)	
Amotosaurus rotfeldensis	SMNS 50691, 50830, 54783, 54784a, 54784b, 54810, 90540, 90543, 90544, 90552, 90559, 90563, 90564, 90566, 90599–90601, several unnumbered specimens; Fraser & Rieppel (2006)	
Macrocnemus bassanii	PIMUZ T2472, T4355, T4822; BSPG 1973 I 86 (cast of Besano II); Peyer (1937); Rieppel (1989a)	
Tanystropheus longobardicus	PIMUZ T2189, T2793, T2817, T2818, T3901; SMNS 54147, 54626, 54628, 54630, 54631, 54632, 54654, 55341, 56289, 59380, 84821, SMNS unnumbered specimen; Wild (1973); Nosotti (2007)	
Jesairosaurus lehmani	ZAR 06–15; Jalil (1997)	
Azendohsaurus madagaskarensis	FMNH PR 2751; UA 7-20-99-653, 8-7-98-284, 8-22-97-91, 8-27-98-273, 8-29-97-151, 8-29-97-152, 8-29-97-160, 10603, 10604, UA unnumbered specimens; Flynn et al. (2010); Nesbitt et al. (2015)	
Pamelaria dolichotrachela	ISI R316–333; Sen (2003)	
Trilophosaurus buettneri	USNM mounted skeleton; Gregory (1945); Parks (1969); Spielmann et al. (2008)	
Noteosuchus colletti	AM 3591; Carroll (1976)	
Mesosuchus browni	SAM-PK-5861, 5882, 6046, 6536, 7416, 7838; Haughton (1921); Carroll (1976); Dilkes (1998)	
Howesia browni	NHMUK PV R5872 (cast), SAM-PK-5884-5886; Broom (1906); Carroll (1976); Dilkes (1995)	
Eohyosaurus wolvaardti	SAM-PK-K10159; Butler et al. (2015)	
Rhynchosaurus articeps	BRLSI M20a, b; NHMUK PV R1236–41; SHYMS 1–7, G3851, G07537; Benton (1990)	
Bentonyx sidensis	BSPG 3D print of BRSUG 21200; unpublished pictures; Hone & Benton (2008); Langer et al. (2010a)	
Eorasaurus olsoni	PIN 156/108–110; Sennikov (1997); Ezcurra, Scheyer & Butler (2014)	
Prolacertoides jimusarensis	IVPP V3233; Young (1973a)	
Prolacerta broomi	BP/1/471, 2675, 2676, 4504a, 5066, 5375; GHG 431; SAM-PK-K10018, K10797; UMCZ 2003.41R; Gow (1975); Evans (1986); Modesto & Sues (2004)	
Kadimakara australiensis holotype	QM F6710; Bartholomai (1979)	
Kadimakara australiensis combined	QM F6676, F6710; Bartholomai (1979)	
Boreopricea funerea	PIN 3708/1; Tatarinov (1978); Benton & Allen (1997)	
Archosaurus rossicus	PIN 1100/55; Tatarinov (1960); Ezcurra, Scheyer & Butler (2014)	
Proterosuchus fergusi holotype	SAM-PK-591; Broom (1903a); Welman (1998); Ezcurra & Butler (2015a)	
Proterosuchus fergusi	BP/1/3993, 4016, 4224; BSPG 1934 VIII 514; GHG 231, 363; RC 59, 846; SAM-PK-11208, K140, K10603; TM 201; Broom (1903a); Broom (1946); Haughton (1924); Broili & Schröder (1934); Cruickshank (1972), Cruickshank (1979); Welman (1998); Ezcurra & Butler (2015a); Ezcurra & Butler (2015b)	
Proterosuchus goweri	NMQR 880; Brink (1955); Ezcurra & Butler (2015a)	
Proterosuchus alexanderi	NMQR 1484; Hoffman (1965); Sereno (1991); Welman (1998); Klembara & Welman (2009); Ezcurra & Butler (2015a)	
“Chasmatosaurus” yuani	IVPP V2719, V4067, V36315 (field number V90002); Young (1936); Young (1963); Young (1978)	
“Chasmatosaurus ultimus”	IVPP V2301; Young (1958); Liu et al. (2015)	
“Ankistrodon indicus”	GSI 2259; Huxley (1865)	
Tasmaniosaurus triassicus	UTGD 54655; Camp & Banks (1978); Thulborn (1986); Ezcurra (2014)	
“Exilisuchus tubercularis”	PIN 4171/25; Ochev (1979)	
“Blomosuchus georgii”	PIN 1025/14, 348; Ochev (1978); Sennikov (1992)	
Vonhuenia friedrichi	PIN 1025/11, 419; Sennikov (1992)	
Chasmatosuchus rossicus	PIN 2252/381, 3200/212, 217, 472, 2243/167, 2252/384, 386; Huene (1940)	
Chasmatosuchus magnus	PIN 951/65; Ochev (1979)	
“Gamosaurus lozovskii”	PIN 3361/13, 14, 94, 183, 213, 214; Ochev (1979); Sennikov (1995b)	
C. magnus+ “G. lozovskii”	PIN 951/65, 3361/13, 14, 94, 183, 213, 214; Ochev (1979); Sennikov (1995b)	
“Chasmatosuchus” vjushkovi	PIN 2394/4; Ochev (1961)	
SAM P41754 Long Reef	SAM P41754; Kear (2009)	
Koilamasuchus gonzalezdiazi	MACN-Pv 18119; Bonaparte (1981); Ezcurra, Lecuona & Martinelli (2010)	
Kalisuchus rewanensis	QM F8998; Thulborn (1979)	
Fugusuchus hejiapanensis	Unpublished photographs; Cheng (1980); Parrish (1992); Gower & Sennikov (1996)	
Sarmatosuchus otschevi	PIN 2865/68; Sennikov (1994); Gower & Sennikov (1997)	
Guchengosuchus shiguaiensis	IVPP V8808; Peng (1991)	
Cuyosuchus huenei	MCNAM PV 2669; Rusconi (1951); Desojo, Arcucci & Marsicano (2002)	
GHG 7433MI	GHG 7433MI	
Garjainia prima	PIN 2394/5, 951/8, 15, 23, 25, 27–30, 32–34, 36, 41, 42, 46, 50, 51, 54, 55, 63; Ochev (1958); Huene (1960); Gower (1996); Gower & Sennikov (1996)	
Garjainia madiba holotype	BP/1/5760; Gower et al. (2014)	
Garjainia madiba combined	BP/1/5760, and BP and NMQR specimens reported by Gower et al. (2014); Gower et al. (2014)	
Erythrosuchus africanus	AMNH 5594–5597, 19352, 19353; BP/1/2094, 2096, 2529, 2734, 3893, 4526, 4539, 4553, 4645, 4649, 4680, 5207; GHG AK-82-22; MNHN 1869-12; NHMUK PV R525, R533i, R2790, R3301, R3592, R3762–R3764, unnumbered specimens; NMQR 1473; SAM-PK-905, 912, 913, 930, 978, 1315, 3028, 3612, 7684, 11330, K1098, K1118, K10024, K10025; UMCZ T666, T700; Broom (1906); Brink (1955); Cruickshank (1978); Parrish (1992); Gower (1996); Gower (1997); Gower (2001); Gower (2003)	
Shansisuchus shansisuchus	IVPP V2501, V2504, V2505, unnumbered specimens; Young (1964); Parrish (1992); Gower (1996); Gower & Sennikov (1996)	
Shansisuchus kuyeheensis	Cheng (1980)	
Chalishevia cothurnata	PIN 2867/7, 18, 4366/1, 2, 3, 8; Ochev (1980); Sennikov (1995a); Sennikov (1995b); Gower & Sennikov (2000)	
Youngosuchus sinensis	IVPP V3239; Young (1973b)	
“Dongusia colorata”	PIN 268/2; Huene (1940)	
Uralosaurus magnus holotype	PIN 2973/70; Ochev (1980)	
Uralosaurusmagnus combined	PIN 2973/70–79; Ochev (1980)	
Vancleavea campi	AMNH 30884 (casts); USNM 508519 (cast of GR 138); Long & Murry (1995); Hunt, Lucas & Spielmann (2005); Parker & Barton (2008); Nesbitt et al. (2009)	
Asperoris mnyama	NHMUK PV R36615; Nesbitt et al. (2013)	
Euparkeria capensis	AMNH 2238, 2239, 5548, 19351; GPIT 1681/11; SAM-PK-1100, 3427, 5867, 5883, 6047A, 6047B, 6048–6050, 6557, 7411, 7659, 7696, 7700, 7702–7713, 7868, 10011, 10671, 13664–13667, K335, K8050, K8051, K8309, K10010, K10012, K10548; UMCZ T692; Broom (1903a); Haughton (1921); Ewer (1965); Gow (1970); Cruickshank (1979); Gower & Weber (1998); Senter (2003)	
Dorosuchus neoetus	PIN 952/200, 1579/61–68; Sennikov (1989a); Sennikov (1989b); Sennikov (1995b); Sennikov (2008); Sookias et al. (2014a)	
Yarasuchus deccanensis	ISI R334; Sen (2005)	
Dongusuchus efremovi	PIN 952/15-1–5; Sennikov (1988a); Sennikov (1988b); Sennikov (1990); Sennikov (1995b); Niedźwiedzki, Sennikov & Brusatte (2014)	
Proterochampsa barrionuevoi	MACN-Pv 18165; PVL 2058, 2061, 2063; PVSJ 606; Reig (1959); Sill (1967); Trotteyn & Haro (2011); Trotteyn (2011a); Dilkes & Arcucci (2012)	
Proterochampsa nodosa	MCP 1694; Barberena (1982); Dilkes & Arcucci (2012)	
Tropidosuchus romeri	PVL 4601, 4602, 4604, 4606; Arcucci (1990)	
Cerritosaurus binsfeldi	UFRGS cast of CA s/n; unpublished photographs; Price (1946); Trotteyn, Arcucci & Raugust (2013)	
Gualosuchus reigi	PULR 05; PVL 4576; Romer (1971a); Dilkes & Arcucci (2012); Trotteyn, Arcucci & Raugust (2013)	
Chanaresuchus bonapartei	MCZ 4035, 4037, 4039; PULR 07; PVL 4575, 4586, 4647, 4676, 6244; Romer (1971a); Romer (1972b); Cruickshank (1979); Sereno (1991); Trotteyn & Haro (2012)	
Pseudochampsa ischigualastensis	PVSJ 567; Trotteyn & Haro (2012); Trotteyn, Martínez & Alcober (2012); Trotteyn & Ezcurra (2014)	
Rhadinosuchus gracilis	BSPG AS XXV 50, 51; Huene (1938); Ezcurra, Desojo & Rauhut (2015)	
Archeopelta arborensis	CPEZ-239a; Desojo, Ezcurra & Schultz (2011)	
Tarjadia ruthae	MCZ 4076, 4077, 9319, unnumbered specimen; PULR 063; Arcucci & Marsicano (1998)	
Jaxtasuchus salomoni	SMNS 91002, 91083, 91352; Schoch & Sues (2014)	
Doswellia kaltenbachi	USNM 25840, 186989, 214823, 244214; Weems (1980); Long & Murry (1995); Dilkes & Sues (2009)	
Parasuchus hislopi	ISI R42–44; Chatterjee (1978); Kammerer et al. (2015)	
Parasuchus agustifrons	BSPG 1931 X 502; Butler et al. (2014b)	
Nicrosaurus kapffi	NHMUK PV R38036, 38037, 42743–42745; SMNS multiple specimens; Huene (1923); Hungerbühler (1998); Hungerbühler (2000)	
Smilosuchus spp.	MCZ 1029; UCMP 26699, 27200; USNM 18313; Camp (1930); Sereno (1991)	
Ornithosuchus longidens	NHMUK PV R2409, R2410, R3142, R3143, R3149, R3562, R3622, R3916; Huene (1914); Walker (1964); Sereno (1991)	
Riojasuchus tenuisceps	PVL 3814, 3826–3828; Bonaparte (1971)	
Nundasuchus songeaensis	Nesbitt et al. (2014)	
Turfanosuchus dabanensis	IVPP V3237; Young (1973c); Wu & Russell (2001); Butler et al. (2014a)	
Gracilisuchus stipanicicorum	CRILAR-Pv 490; MCZ 4116–4118; PULR 08; PVL 4597, 4612; Romer (1972c); Lecuona & Desojo (2011); Butler et al. (2014a)	
Aetosauroides scagliai	MCP 13a-b-PV; PVL 2052, 2059, 2073; UFSM 11070a; Casamiquela (1960); Casamiquela (1961); Casamiquela (1967); Desojo & Ezcurra (2011)	
Batrachotomus kupferzellensis	SMNS 52970, 80260, 80269, 80271, 80273, 80275, 80276, 80280, 80283, 80285, 80288, 80293, 80294, 80296, 80300, 80305, 80309, 80322, 91044, 91049, cast of MHI 1895; Gower (1999); Gower & Schoch (2009)	
Prestosuchus chiniquensis	UFRGS-PV-0137-T, 0152-T, 0156-T, 0473-T, 0629-T; Barberena (1978); Nesbitt (2011); Mastrantonio et al. (2013)	
Dimorphodon macronyx	NHMUK PV R41212-13, R1034, R1035; Padian (1983); Nesbitt (2011)	
Lagerpeton chanarensis	MCZ 4121; PULR 06; PVL 4619, 4625; Romer (1971b); Romer (1972e); Arcucci (1986); Sereno & Arcucci (1994a)	
Marasuchus lilloensis	PVL 3870–3872, 4671, 4672; Romer (1971b); Romer (1972e); Bonaparte (1975); Sereno & Arcucci (1994b)	
Lewisuchus admixtus	PULR 01; Romer (1972d); Bittencourt et al. (2014)	
Silesaurus opolensis	Multiple ZPAL specimens; Dzik (2003); Dzik & Sulej (2007); Piechowski & Dzik (2010); Langer et al. (2013); Kubo & Kubo (2014); Piechowski, Tałanda & Dzik (2014)	
Heterodontosaurus tucki	AM 4765, unnumbered specimen; SAM-PK-K337, K1332; Crompton & Charig (1962); Santa Luca (1980); Norman et al. (2011); Porro et al. (2011); Sereno (2012)	
Herrerasaurus ischigualastensis	MACN-Pv 18060; MCZ 4381, 7064; MLP 61-VIII-2-2, 61-VIII-2-3; PVL 2054, 2558, 2566; PVSJ 053, 104, 373, 380, 407; Reig (1963); Brinkman & Sues (1987); Novas (1986); Sereno & Novas (1994); Novas (1992a); Novas (1993); Sereno (1994); Ezcurra (2010a); Ezcurra (2012)	

Table 2 Percentages of missing entries and polymorphisms present in the operational taxonomic units of the phylogenetic analysis.

Terminals in bold were removed a posteriori to generate the series of strict reduced consensus trees and terminals indicated with an asterisk were removed a posteriori from the second round of decay indices.

Terminal	Miss. ent.	Non-app.	Anat. poly	Amb. poly	
Petrolacosaurus kansensis	10.67	10.83	0.33	0.00	
Acerosodontosaurus piveteaui*	66.17	5.33	0.00	0.00	
Youngina capensis	8.33	8.83	0.83	0.00	
Paliguana whitei*	84.83	2.83	0.00	0.00	
Planocephalosaurus robinsonae	32.33	8.83	0.17	0.00	
Gephyrosaurus bridensis	28.50	7.00	0.17	0.5	
Cteniogenys sp.	54.83	4.83	0.00	0.67	
Simoedosaurus lemoinei	20.83	7.83	0.50	0.17	
Aenigmastropheus parringtoni*	94.50	0.00	0.00	0.17	
Protorosaurus speneri	25.50	7.17	1.00	0.83	
Amotosaurus rotfeldensis	50.67	5.67	0.50	0.33	
Macrocnemus bassanii	33.17	8.17	0.67	0.00	
Tanystropheus longobardicus	10.83	11.67	1.67	0.50	
Jesairosaurus lehmani	49.17	8.00	0.67	0.83	
Pamelaria dolichotrachela	22.17	6.50	0.17	0.67	
Azendohsaurus madagaskarensis	8.83	7.33	0.67	0.17	
Trilophosaurus buettneri	12.17	9.83	0.17	0.67	
Noteosuchus colletti*	74.00	2.00	0.17	0.33	
Mesosuchus browni	11.33	7.00	0.17	0.17	
Howesia browni	46.33	4.67	0.17	0.83	
Eohyosaurus wolvaardti*	78.67	1.50	0.00	0.67	
Rhynchosaurus articeps	23.17	8.50	0.33	0.17	
Bentonyx sidensis*	64.33	5.83	0.00	0.17	
Eorasaurus olsoni*	96.17	0.00	0.17	0.17	
Prolacertoides jimusarensis*	87.33	2.00	0.00	0.17	
Prolacerta broomi	4.67	7.00	1.50	0.33	
Kadimakara australiensis holotype*	88.50	0.50	0.00	0.50	
Kadimakara australiensis combined*	84.17	2.50	0.00	0.83	
Boreopricea funerea	53.67	4.83	0.00	0.83	
Archosaurus rossicus*	97.00	0.00	0.00	0.00	
Proterosuchus fergusiholotype*	91.33	0.17	0.00	0.67	
Proterosuchus fergusi	22.50	3.33	2.17	0.17	
Proterosuchus goweri*	63.17	1.00	0.00	0.50	
Proterosuchus alexanderi	28.50	3.83	0.00	0.50	
“Chasmatosaurus” yuani	22.67	4.83	0.83	1.33	
“Chasmatosaurus ultimus”*	91.17	0.17	0.00	0.17	
“Ankistrodon indicus”*	98.83	0.00	0.00	0.17	
Tasmaniosaurus triassicus*	85.17	0.50	0.00	1.00	
“Exilisuchus tubercularis”*	98.50	0.00	0.00	0.00	
“Blomosuchus georgii”*	97.67	0.00	0.00	0.00	
Vonhuenia friedrichi*	96.83	0.33	0.00	0.00	
Chasmatosuchus rossicus*	93.83	0.33	0.17	0.00	
Chasmatosuchus magnus*	97.17	0.17	0.00	0.00	
“Gamosaurus lozovskii”*	97.00	0.00	0.00	0.00	
C. magnus+G. lozovskii*	96.17	0.17	0.00	0.00	
“Chasmatosuchus”vjushkovi*	96.83	0.00	0.00	0.33	
SAM P41754 Long Reef*	97.17	0.17	0.00	0.17	
Koilamasuchus gonzalezdiazi*	95.00	0.33	0.00	0.50	
Kalisuchus rewanensis*	94.67	0.33	0.00	0.67	
Fugusuchus hejiapanensis*	70.33	1.83	0.00	0.33	
Sarmatosuchus otschevi*	65.83	1.00	0.17	1.00	
Guchengosuchus shiguaiensis*	71.83	2.83	0.00	1.17	
Cuyosuchus huenei*	73.67	0.83	0.17	1.00	
GHG 7433MI*	77.33	2.00	0.00	0.50	
Garjainia prima	17.33	4.00	0.33	0.33	
Garjainia madiba holotype*	94.33	0.33	0.00	0.17	
Garjainia madiba combined	52.50	1.33	0.50	1.17	
Erythrosuchus africanus	9.67	5.33	0.67	0.00	
Shansisuchus shansisuchus	24.83	5.00	0.33	0.50	
Shansisuchus kuyeheensis*	88.00	0.33	0.00	0.50	
Chalishevia cothurnata*	86.33	0.33	0.00	0.33	
Youngosuchus sinensis	52.33	2.17	0.00	0.83	
“Dongusia colorata”*	97.33	0.00	0.00	0.00	
Uralosaurus magnusholotype*	98.00	0.83	0.00	0.00	
Uralosaurusmagnuscombined*	95.33	0.83	0.00	0.00	
Vancleavea campi	31.67	7.50	0.17	0.33	
Asperoris mnyama*	84.50	1.33	0.00	1.00	
Euparkeria capensis	6.83	2.67	1.83	0.17	
Dorosuchus neoetus*	85.00	0.17	0.00	0.00	
Yarasuchus deccanensis*	67.83	1.50	0.17	0.83	
Dongusuchus efremovi*	95.83	0.00	0.00	0.00	
Proterochampsa barrionuevoi	31.33	6.17	0.33	1.00	
Proterochampsa nodosa*	69.33	1.67	0.00	0.17	
Tropidosuchus romeri	24.33	3.67	0.50	0.83	
Cerritosaurus binsfeldi*	68.00	2.33	0.17	0.33	
Gualosuchus reigi	30.33	4.00	0.67	0.50	
Chanaresuchus bonapartei	11.33	4.83	1.67	0.00	
Pseudochampsa ischigualastensis	42.33	2.83	0.00	0.83	
Rhadinosuchus gracilis*	85.00	0.50	0.17	0.83	
Archeopelta arborensis*	82.17	0.67	0.00	0.17	
Tarjadia ruthae*	90.33	0.17	0.00	0.17	
Jaxtasuchus salomoni*	71.67	1.00	0.17	1.00	
Doswellia kaltenbachi	48.67	5.33	0.33	0.67	
Parasuchus hislopi	12.83	4.00	0.33	0.33	
Parasuchus angustifrons*	63.67	3.83	0.00	0.33	
Nicrosaurus kapffi	35.67	5.00	0.67	0.50	
Smilosuchus spp.	12.33	5.83	0.50	0.17	
Ornithosuchus longidens	32.00	3.00	0.17	0.33	
Riojasuchus tenuisceps	14.33	5.33	0.50	0.50	
Nundasuchus songeaensis*	62.83	1.33	0.17	0.17	
Turfanosuchus dabanensis	33.17	2.33	0.00	0.83	
Gracilisuchus stipanicicorum	20.83	4.00	0.67	1.00	
Aetosauroides scagliai	38.67	2.33	0.50	1.50	
Batrachotomus kupferzellensis	19.67	3.50	0.33	0.33	
Prestosuchus chiniquensis	10.33	3.50	0.50	0.33	
Dimorphodon macronyx	45.83	7.17	0.33	0.33	
Lagerpeton chanarensis*	76.00	2.67	0.17	0.00	
Marasuchus lilloensis	49.50	3.50	0.17	0.67	
Lewisuchus admixtus	50.83	2.33	0.17	0.83	
Silesaurus opolensis	18.50	6.50	0.83	0.67	
Heterodontosaurus tucki	10.67	10.00	0.33	0.00	
Herrerasaurus ischigualastensis	9.83	6.83	1.67	0.33	
Mean (standard deviation)	55.94 (±31.93)	3.34 (±2.98)	0.29 (±0.44)	0.42 (±0.36)	
Median	58.83	2.67	0.17	0.33	
Notes.

Amb. poly. polymorphisms due to ambiguities in scorings

Anat. poly. anatomical polymorphisms

Miss.ent. missing entries

Non-app. non-applicable characters

The characters were scored in Mesquite 3.01 (Maddison & Maddison, 2015) based on first-hand observations in more than 95% of the sampled terminals (Tables 1 and 2). Only five terminals were scored entirely based on published information: Petrolacosaurus kansensis, Gephyrosaurus bridensis, Fugusuchus hejiapanensis, Shansisuchus kuyeheensis, and Nundasuchus songeaensis. The holotypes and only known specimens of Fugusuchus hejiapanensis and Shansisuchus kuyeheensis could not be studied at first-hand because they could not be located by the GMB and IGCAGS when access was requested in May 2013. Scorings were treated as polymorphic if a character-state is present in some individuals of a terminal but not in others, and the following notation was used: state A & state B. If a character-state could not be determined because of lack of preservation but other states can be ruled out based on positive evidence, the scoring was also treated as polymorphic but using the following notation: state A/state C. Characters were scored based on osteologically mature or the largest known individuals when extensive growth series are available (e.g., Tanystropheus longobardicus, Proterosuchus fergusi).

Character descriptions

The characters used in the current phylogenetic analysis are listed and described as follows. Several of these characters were described in detail by previous authors (e.g., Nesbitt, 2011; Pritchard et al., 2015) and in most of these cases a new discussion was not considered necessary.

Cranium

1. Skull and lower jaws, interdental plates: absent (0); present, but restricted to the anterior end of the dentary (1); present along the entire alveolar margin of the premaxilla, maxilla and dentary (2) (modified from Carrano, Sampson & Forster, 2002; Pritchard et al., 2015: 96, in part), ORDERED (Figs. 13 and 14).

Figure 14 Teeth and tooth implantation in Triassic and Early Jurassic archosauromorphs in (A, C, E, G, H, K, M) labial, (B, D, F) mesial, (I) occlusal, and (J, L) lingual views.

Isolated teeth of (A, B) Tanystropheus longobardicus (SMNS 54147), (C, D) Azendohsaurus laaroussi (MNHN-ALM 424) and (E, F) Chalishevia cothurnata (PIN 4366/8), and (G) teeth in situ of Prolacerta broomi (BP/1/4504a), (H) Azendohsaurus madagaskarensis (UA 8-29-97-160), (I, J) indeterminate rhynchosaurid from the Middle Triassic of England (WARMS G955), (K) Proterosuchus fergusi (BSPG 1934-VIII-514, mirrored), (L) Erythrosuchus africanus (BP/1/2529, mirrored) and (M) Heterodontosaurus tucki (SAM-PK-K1332). Close-ups of ankylothecodont tooth implantation in (K) and mesial denticles in (L). Numbers indicate character-states scored in the data matrix and the arrows indicate anterior direction. Scale bars equal 2 mm in (A, B, I, J), 1 mm in (C, D, G), close-up of (K), close-up of (L), 5 mm in (E, F, H), 2 cm in (K, L), and 1 cm in (M).

The interdental plates (=paradental plates) contribute to the medial wall of the alveoli of the marginal dentition in most archosauriforms (Nesbitt, 2011), but they are apomorphically lost in some archosaur groups, such as sauropodomorphs and crocodiles (e.g., Panphagia protos: PVSJ 874; Caiman latirostris: MACN-He 43694). The interdental plates are restricted to the anterior end of the dentary in “Chasmatosaurus” yuani (IVPP V36315), a specimen previously referred to “Chasmatosaurus” sp. from the Early Triassic of India (GSI 18124), and an isolated dentary previously referred to Archosaurus rossicus (PIN 1100/78). In these taxa, the interdental plates are low and their bases are broadly separated from each other. By contrast, the interdental plates are present along the entire alveolar margin of the skull and lower jaw, broader at their bases in erythrosuchids and also higher in more crownward archosauriforms (e.g., Euparkeria capensis: UMZC T692; Batrachotomus kupferzellensis: SMNS 52970). As a result, the presence of interdental plates restricted to the anterior end of the dentary was added here as a third state to the historically binary character of absence/presence of these structures (e.g., Carrano, Sampson & Forster, 2002). Nesbitt (2011) stated that the origin of interdental plates in archosauriforms seems to coincide with the appearance of a thecodont tooth implantation. However, it was observed here that some proterosuchian archosauriforms possess interdental plates, but retain at the same time an ankylothecodont tooh implantation, as is the case in Kalisuchus rewanensis (QM F8998) and the anterior end of the dentary of “Chasmatosaurus” yuani (IVPP V36315) and “Chasmatosaurus” sp. from India (GSI 18124).

2. Skull, total length versus length of the presacral vertebral column: 0.22–0.38 (0); 0.44–0.72 (1); 0.94–0.98 (2) (modified from Sereno, 1991; Ezcurra, Lecuona & Martinelli, 2010: 113; Nesbitt, 2011: 134), ORDERED. This character is inapplicable in taxa with an extremely elongated neck (e.g., Tanystropheus longobardicus).

See comments in Nesbitt (2011: character 134) (Figs. 15 and 22).

Figure 15 Partial skeletons of Permo-Triassic neodiapsids.

(A) Youngina capensis in left lateral view (SAM-PK-K8565, mirrored), (B) Macrocnemus bassanii (PIMUZ T4355), (C) Tanystropheus longobardicus (PIMUZ T2817, mirrored), (D) Jesairosaurus lehmani in right lateral view (ZAR 06), (E) Euparkeria capensis (SAM-PK-5867), (F) Pseudochampsa ischigualastensis in dorsal view (PVSJ 567), and (G) Proterosuchus alexanderi in left lateral view (NMQR 1484, mirrored). Numbers indicate character-states scored in the data matrix and the arrows indicate anterior direction. Scale bars equal 1 cm in (A, D), 2 cm in (B), 20 cm in (C), and 5 cm in (E–G).

3. Skull, strongly dorsoventrally compressed skull with mainly dorsally facing antorbital fenestrae and orbits: absent (0); present (1) (Reig, 1959; Trotteyn & Ezcurra, 2014: 104).

The skull of some saurian diapsids, including choristoderans (Sigogneau-Russell & Russell, 1978; Evans, 1990), some proterochampsids (Romer, 1971a; Dilkes & Arcucci, 2012; Trotteyn, Arcucci & Raugust, 2013), Doswellia kaltenbachi (Weems, 1980; Dilkes & Sues, 2009) and phytosaurs (Butler et al., 2014b; Stocker & Butler, 2013) is strongly dorsoventrally compressed and, as a result, the external nares, antorbital fenestrae (if present) and orbits face mainly dorsally. The presence of a compressed skull may be related with a semi-aquatic or aquatic mode of life (Fig. 16).

Figure 16 Skulls of Triassic archosauromorphs in (A, C–E) dorsal and (B) left lateral views.

(A, B) Bentonyx sidensis (BRSUG 27200); (C) Chanaresuchus bonapartei (PULR 07); (D) Protorosaurus speneri (MCP 1694); and (E) Proterosuchus fergusi (BP/1/4016). Close up of the pair of septomaxillae in dorsal view in (E). Numbers indicate character-states scored in the data matrix and anterior direction is towards the left. Scale bars equal 2 cm in (A–C, E) and 5 cm in (D).

4. Skull, well-developed nodular prominences on the lateral surface of maxilla, jugal, quadratojugal, squamosal and angular: absent (0); present (1) (Sill, 1967; Trotteyn & Ezcurra, 2014: 105) (Fig. 16).

The skull of Proterochampsa nodosa and some specimens of Proterochampsa barrionuevoi possess a series of nodular prominences on its lateral surface (Sill, 1967; Barberena, 1982; Dilkes & Arcucci, 2012). This feature seems to be uniquely shared by these species among Permo-Triassic archosauromorphs.

5. Skull, dermal sculpturing on the dorsal surface of the skull roof: absent (0); shallow or deep pits scattered across surface and/or low ridges (1); prominent ridges or tubercles on frontals, parietals, and nasals (2) (Dilkes & Arcucci, 2009; modified from Dilkes & Arcucci, 2012: 1) (Fig. 16).

The skull roof of multiple diapsids possesses an ornamented dorsal surface, being represented by shallow or deep pits of random arrangement and/or low ridges in disparate groups (e.g., Petrolacosaurus kansensis: Reisz, 1981: Fig. 8A; Gephyrosaurus bridensis: Evans, 1980; Cteniogenys sp.: Evans, 1990; Amotosaurus rotfeldensis: SMNS 54783b; Proterosuchus alexanderi: NMQR 1484; Vancleavea campi: Nesbitt et al., 2009; Asperoris mnyama: Nesbitt, Butler & Gower, 2013; Nicrosaurus kapffi: NHMUK PV R42743; Gracilisuchus stipanicicorum: MCZ 4117), or prominent ridges or tubercles in proterochampsids Romer, 1971a; Trotteyn, Arcucci & Raugust, 2013, doswelliids (Weems, 1980; MCZ 9319; USNM 214823), and the phytosaur genus Parasuchus (Butler et al., 2014b; BSPG 1931 X 502; ISI R42). The presence of pits and/or low ridges is probably a highly homoplastic character, but it may be informative at low phylogenetic levels.

6. Skull, dorsal surface of nasals and/or frontals ornamented by ridges radiating from centres of growth: absent (0); present (1) (Romer, 1971a; Trotteyn & Ezcurra, 2014: 106). This character is inapplicable to taxa that lack ridges or tubercles on the dorsal surface of the skull roof (i.e., character-state 5-0) (Fig. 16).

The ornamentation of the skull roof is arranged in a radial pattern from a centre of growth, rather than in a longitudinal or random arrangement, in the early diapsid Petrolacosaurus kansensis (Reisz, 1981), the lepidosauromorph Gephyrosaurus bridensis (Evans, 1980), and some proterochampsids (Gualosuchus reigi, Chanaresuchus bonapartei, Pseudochampsa ischigualastensis and Rhadinosuchus gracilis; (Romer, 1971a; Trotteyn, Martínez & Alcober, 2012; Trotteyn, Arcucci & Raugust, 2013; Trotteyn & Ezcurra, 2014; Ezcurra, Desojo & Rauhut, 2015).

7. Skull, dorsal orbital margin: orbital margin of the frontal level with skull table or raised slightly (0); orbital margin of the frontal elevated above skull Table 1; shelf/ridge elevated above skull table and extends along the lateral surface of the lacrimal, prefrontal, frontal portion of orbital rim, and postorbital (2) (Dilkes & Sues, 2009; Ezcurra, Lecuona & Martinelli, 2010: 16; modified from Dilkes & Arcucci, 2012: 15, 20), ORDERED (Figs. 16 and 18).

The dorsal border of the orbit of proterochampsids is raised above the skull table as part of a ridge that extends along the lateral surface of the lacrimal, prefrontal, frontal and postorbital (Dilkes & Arcucci, 2012). By contrast, the dorsal border of the orbit of phytosaurs and some other archosaurs is elevated above the skull roof, but it does not form part of a continuous longitudinal ridge on the dorsolateral surface of the skull. In other terminals, the dorsal border of the orbit is at level with the skull roof.

8. Skull, dorsal surface of the temporal region: flat (0); supratemporal fossa immediately medial or anterior to the supratemporal fenestra (1); thin, blade-like median sagittal crest (2) (Laurin, 1991: G2; Müller, 2004: 13, in part; Reisz, Laurin & Marjanović, 2010: 30; Nesbitt, 2011: 59; Dilkes & Arcucci, 2012; Ezcurra, Scheyer & Butler, 2014: 29, in part; Pritchard et al., 2015: 20). This character is inapplicable in taxa lacking supratemporal fenestrae (Figs. 16 and 23).

See comments in Nesbitt (2011: character 59).

9. External nares, confluent: absent (0); present (1) (Laurin, 1991: F2; Müller, 2004: 85; Reisz, Laurin & Marjanović, 2010: 20; Ezcurra, Lecuona & Martinelli, 2010: 5; modified from Ezcurra, Scheyer & Butler, 2014: 20; Pritchard et al., 2015: 3), ORDERED (Figs. 16, 17 and 20).

Figure 17 Reconstructions of the skull and lower jaw of Permo-Triassic archosauromorphs in lateral view.

(A) Protorosaurus speneri; (B) Azendohsaurus madagaskarensis; (C) Mesosuchus browni; (D) Prolacerta broomi; (E) Proterosuchus fergusi; and (F) Garjainia prima. Numbers indicate character-states scored in the data matrix and anterior direction in towards the right. Scale bars equal 1 cm in (A–D), and 5 cm in (E, F). (A) Modified from Gottmann-Quesada & Sander (2009), (B) modified from Flynn et al. (2010), and (C) modified from Dilkes (1998).

The external nares of choristoderans, allokotosaurians and rhynchosaurs are confluent along the midline due to the absence of a prenarial process of the premaxilla (Sigogneau-Russell & Russell, 1978; Evans, 1990; Dilkes, 1998; Flynn et al., 2010). By contrast, in the vast majority of diapsids, the prenarial process of the premaxilla contacts the nasal, forming a complete internarial bar.

10. External naris, anteroposterior position in the snout: terminal, on the anterior end of the snout (0); nonterminal, considerably posteriorly displaced, but posterior rim of the naris well anterior to the anterior border of the orbit (1); nonterminal, considerably posteriorly displaced and posterior rim of the naris approximately at level with the anterior border of the orbit (2) (Dilkes, 1998; Ezcurra, Lecuona & Martinelli, 2010: 5, in part; modified from Nesbitt, 2011: 139), ORDERED (Fig. 18).

Figure 18 Skulls of Triassic and Early Jurassic archosauriforms in lateral view.

(A) Euparkeria capensis (SAM-PK-5867); (B) Vancleavea campi (USNM 508519 [cast of GR 138]); (C) Gualosuchus reigi (PVL 4576); (D) Parasuchus hislopi (ISI R42, mirrored); (E) Gracilisuchus stipanicicorum (MCZ 4117); and (F) Heterodontosaurus tucki (SAM-PK-K337). Numbers indicate character-states scored in the data matrix and anterior direction is towards the right. Scale bars equal 1cm in (A, B, E, F), 2 cm in (C), and 5 cm in (D). Photograph (B) courtesy of Sterling Nesbitt.

See comments in Nesbitt (2011: character 139).

11. External naris, directed: laterally (0); dorsally (1); anteriorly (2) (Sereno, 1991; Nesbitt, 2011: 140; modified from Dilkes & Arcucci, 2012: 6) (Figs. 16 and 18).

See comments in Nesbitt (2011: character 140).

12. External naris, shape: sub-circular (0); oval (1) (Dilkes, 1998: 12; Müller, 2004: 86; Ezcurra, Scheyer & Butler, 2014: 116) (Fig. 19).

Figure 19 Skulls of Early and Middle Triassic archosauriforms in (A) left lateral and (B) right lateral views.

(A) Proterosuchus fergusi (RC 846, mirrored); and (B) Erythrosuchus africanus (BP/1/5207). Numbers indicate character-states scored in the data matrix and anterior direction is towards the right. Scale bars equal 5 cm in (A) and 10 cm in (B).

The scoring of this character through the taxonomic sample of this analysis showed that the presence of a sub-circular external naris in lateral view is scattered present in a few diapsids, including the choristoderan Simoedosaurus lemoinei (Sigogneau-Russell & Russell, 1978), the rhynchocephalian Planocephalosaurus robinsonae (Fraser, 1982) and the archosauriform Euparkeria capensis (SAM-PK-5867).

13. Antorbital fenestra: absent (0); present (1) (Juul, 1994; Dilkes, 1998: 5; Ezcurra, Lecuona & Martinelli, 2010: 3; Nesbitt, 2011: 136; Dilkes & Arcucci, 2012: 2; Ezcurra, Scheyer & Butler, 2014: 118; Pritchard et al., 2015: 13) (Fig. 17).

See comments in Nesbitt (2011: character 136).

14. Antorbital fenestra, anterior margin: gently rounded (0); nearly pointed (1) (Benton & Clark, 1988; Nesbitt, 2011: 30). The character is not applicable to taxa that lack an antorbital fenestra (Fig. 18).

Figure 20 Left (A, D, E) and right (B, C) premaxillae of Permo-Triassic archosauromorphs in lateral view.

(A) Archosaurus rossicus (PIN 1100/55); (B) Sarmatosuchus otschevi (PIN 2865/68-9, mirrored); (C) Azendohsaurus madagaskarensis (UA 8-7-98-284, mirrored); (D) Shansisuchus shansisuchus (IVPP V2505); and (E) Batrachotomus kupferzellensis (SMNS 80260). Numbers indicate character-states scored in the data matrix and the arrows indicate anterior direction. Scale bars equal 1 cm in (A, B, D, E) and 5 mm in (C).

See comments in Nesbitt (2011: character 30).

15. Secondary antorbital fenestra, immediately anterior to the antorbital fenestra: absent (0); present (1) (New character) (Fig. 22).

The anteroventral process of the nasal bifurcates distally to form the dorsal border of a secondary antorbital fenestra, placed between the premaxilla and maxilla, in the erythrosuchids Guchengosuchus shiguaiensis, Shansisuchus shansisuchus and Chalishevia cothurnata (Young, 1964; Ochev, 1980; Peng, 1991; Ezcurra, Butler & Gower, 2013; Wang et al., 2013). This cranial opening differs from and is not considered homologous to the smaller and usually circular or slit-like foramen present between the premaxilla and maxilla of several basal loricatans (e.g., Prestosuchus chiniquensis: Lacerda et al., 2016; Saurosuchus galilei: Alcober, 2000), in which the nasal does not contribute to its border.

16. Orbit, shape: anteroposteriorly longer than tall (0); subcircular (1); dorsoventrally taller than long (2) (Benton & Clark, 1988; modified from Nesbitt, 2011: 142).

See comments in Nesbitt (2011: character 142) (Figs. 16–18).

17. Orbit, elevated rim: absent or incipient (0); present, restricted to the ascending process of the jugal and sometimes also onto the ventral process of the postorbital (1); present, well-developed along the jugal, postorbital, frontal, prefrontal and lacrimal (2) (modified from Butler et al., 2015: 5),ORDERED (Figs. 16 and 17).

The margin of the orbit is not raised in non-archosauromorph diapsids, such as Protorosaurus speneri, Macrocnemus bassanii, Jesairosaurus lehmani, Trilophosaurus buettneri, Vancleavea campi, phytosaurs and dinosauriforms. By contrast, the lateral surface of the ascending process of the jugal possesses a diagonal, anteroventrally-to-posterodorsally oriented tuberosity adjacent to the border of the orbit in other sampled archosauromorphs. This tuberosity sometimes continues along the jugal-postorbital suture and extends onto the posterolateral portion of the ventral process of the postorbial in some archosauromorph species. In particular, this ridge on the lateral surface of the jugal forms part of an elevated rim along the entire orbital margin (forming a “telescopic orbit”), representing state 2 of this character, in the rhynchosaurid rhynchosaurs Rhynchosaurus articeps and Bentonyx sidensis (Butler et al., 2015).

18. Infratemporal fenestra: present (0); absent (1) (Laurin, 1991: A2; Reisz, Laurin & Marjanović, 2010: 40; Ezcurra, Lecuona & Martinelli, 2010: 39; Pritchard et al., 2015: 29; modified from Nesbitt et al., 2015: 29).

The presence of an infratemporal fenestra is part of the construction of the morphologically diapsid skull (Romer, 1956). This condition has been retained by the majority of archosauromorphs, in which the opening is bounded by the postorbital, squamosal, quadratojugal, and jugal (Bever et al., 2015). However, in the allokotosaurian Trilophosaurus buettneri (Gregory, 1945) and the doswelliid Doswellia kaltenbachi (Weems, 1980) the infratemporal fenestra is secondary lost.

19. Posttemporal fenestra, size: larger than or subequal to the supraoccipital (0); smaller than the supraoccipital (1); developed as a small foramen (2); absent (3) (Sereno & Novas, 1994; Dilkes, 1998: 53, 54; Reisz, Laurin & Marjanović, 2010: 59; Nesbitt, 2011: 141, in part; Ezcurra, Scheyer & Butler, 2014: 58), ORDERED (Fig. 27).

See comments in Nesbitt (2011: character 141).

20. Snout, antorbital length (anterior tip of the skull to anterior margin of the orbit) versus total length of the skull: 0.29–0.40 (0); 0.43–0.62 (1); 0.70–0.76 (2) (Dilkes, 1998: 2; modified from Ezcurra, Scheyer & Butler, 2014: 113), ORDERED (Figs. 17 and 18).

The snout (=antorbital length) is moderately long in comparison with the rest of the skull in several non-archosauriform archosauromorphs and it is the general condition of archosauriforms, in which the snout represents approximately half of the total length of the skull (e.g., Proterosuchus fergusi: RC 846; Garjainia prima: PIN 2394/5). By contrast, the snout is short in lepidosauromorphs, rhynchosaurs, and the archosauriform Vancleavea campi. Conversely, phytosaurs possess greatly elongated snouts, with an antorbital length that exceeds 70% of the total length of the skull (Stocker & Butler, 2013).

21. Snout, dorsoventral height at the level of the anterior tip of the maxilla versus dorsoventral height at the level of the anterior border of the orbit: 0.15–0.30 (0); 0.38–0.53 (1); 0.59–0.80 (2); 1.04 (3) (New character), ORDERED (Figs. 17 and 19).

This character describes the shape of the antorbital region of the skull in lateral view. Taxa with a more subtriangular, anteriorly tapering snout possess a lower ratio (e.g., Macrocnemus bassanii: PIMUZ T4822; Proterosuchus goweri: NMQR 880; Gualosuchus reigi: PULR 05), whereas species with a more sub-rectangular snout possess a higher ratio (e.g., Trilophosaurus buettneri: Spielmann et al., 2008; Riojasuchus tenuisceps: PVL 3827).

22. Snout, proportions at the level of the anterior border of the orbit: transversely broader than dorsoventrally tall or subequal (0); dorsoventrally taller than transversely broad (1) (Reisz & Dilkes, 2003: 53; Reisz, Laurin & Marjanović, 2010: 7; modified from Ezcurra, Scheyer & Butler, 2014: 7) (Fig. 16).

The presence of a strongly dorsoventrally compressed skull (character-state 3–1) seems to be a condition partly included (but not correlated) in the first state of this character. However, it was chosen here to sample both characters independently because the strongly dorsoventrally compressed skull of some archosauromorphs (e.g., some proterochampsids) also involves dorsally facing antorbital fenestrae and orbits, which may not be directly related to a transversely borader than dorsoventrally tall skull.

23. Snout, lateral margin of the snout anterior to the prefrontal: formed by the nasal (0); formed by the nasal and maxilla with gently rounded transition along the maxilla from the lateral to dorsal side of rostrum (1); formed by the nasal and maxilla with sharp edge along the maxilla between the lateral and dorsal sides of this bone (=box-like snout of Kischlat, 2000) (2) (Kischlat, 2000; Dilkes & Arcucci, 2012: 11). This character is inapplicable in taxa with an extensive contact between premaxilla and prefrontal (e.g., rhynchosaurids) (Figs. 18 and 19).

The dorsolateral surface of the snout, immediately anterior to the prefrontal is usually formed only by the nasal in archosauromorphs. By contrast, tha maxilla forms the dorsolateral surface of the snout and the nasal is restricted to the dorsal surface of the antorbital region in Vancleavea campi and proterochampsids. The condition in doswelliids cannot be determined based on preserved specimens. In addition, in the rhadinosuchine proterochampsids Gualosuchus reigi, Chanaresuchus bonapartei, Pseudochampsa ischigualastensis, and Rhadinosuchus gracilis there is a distinct change in slope between the lateral and dorsal surfaces of the snout along the maxilla (Dilkes & Arcucci, 2012). Kischlat (2000) described this condition as a box-like snout.

24. premaxilla–maxilla, suture: simple continuous contact (0); notched along the ventral margin (1) (Dilkes, 1998; Ezcurra, Lecuona & Martinelli, 2010: 7) (Figs. 17 and 19).

The alveolar margin of the skull is continuous between the premaxilla and maxilla in the vast majority of archosauromorphs. In rhynchosaurid rhynchosaurs (e.g., Rhynchosaurus articeps: NHMUK PV R1236; Bentonyx sidensis: BRSUG 21200), Prolacerta broomi (BP/1/471), Proterosuchus spp. (RC 846, NMQR 880), “Chasmatosaurus” yuani (IVPP V36315), some erythrosuchids (e.g., Garjania prima: PIN 2394/5), Vancleavea campi (Nesbitt et al., 2009) and Heterodontosaurus tucki (SAM-PK-K337, K1332) there is a distinct gap between the alveolar margins of the premaxilla and maxilla that forms a notch in lateral view. However, this notched margin seems to be the result of different conditions in different groups. The notch is a result of the strongly modified, hook-like premaxilla in rhynchosaurs, the failure of the premaxilla–maxilla suture to reach the level of the alveolar margin of the skull in Prolacerta, proterosuchids, and erythrosuchids, and the development of a notch to receive a caniniform tooth in the lower jaw of Vancleavea campi and Heterodontosaurus tucki. These apparently different conditions were scored within the same primary-homology hypothesis to test their different phylogenetic origins.

25. premaxilla–maxilla, subnarial foramen between the elements: absent (0); present and the border of the foramen is present on both the maxilla and the premaxilla (1); present and the border of the foramen is present on the maxilla but not on the premaxilla (2) (Benton & Clark, 1988; Nesbitt, 2011: 12) (Figs. 14, 17 and 19 of Nesbitt, 2011).

See comments in Nesbitt (2011: character 12).

26. Premaxilla, alveolar margin does not reach the contact with the maxilla and forms a diastema (=subnarial gap): absent (0); present (1) (Nicholls, 1999; Müller, 2004: 116; Nesbitt, 2011: 11; Ezcurra, Scheyer & Butler, 2014: 117). This character is considered inapplicable in taxa without premaxillary teeth (e.g., Trilophosaurus) (Fig. 17).

See comments in Nesbitt (2011: character 11).

27. Premaxilla, main body size: small, the premaxillary body forms less than half of snout in front of the posterior border of the external nares (0); large, the premaxillary body forms half or more than half of snout in front of the posterior border of the external nares (1) (Rieppel, Mazin & Tchernov, 1999; Müller, 2004: 1; Ezcurra, Scheyer & Butler, 2014: 115) (Fig. 17).

The premaxilla of early diapsids (e.g., Petrolacosaurus kansensis: Reisz, 1981; Youngina capensis: Gow, 1975), basal lepidosauromorphs (e.g., Planocephalosaurus robinsonae: Fraser, 1982; Gephyrosaurus bridensis: Evans, 1980) and the Permian archosauromorph Protorosaurus speneri (Gottmann-Quesada & Sander, 2009) is proportionally very small in comparison with the rest of the skull and forms less than half of the snout in front of the external naris. By contrast, in the choristoderan Simoedosaurus lemoinei Sigogneau-Russell & Russell, 1978 and the vast majority of archosauromorphs, the premaxilla is the main component of the snout in lateral view and forms most or all of the ventral border of the external naris.

28. Premaxilla, anteroposterior length of the main body versus its maximum dorsoventral height: 0.70–0.73 (0); 1.07–2.00 (1); 2.22–3.80 (2); 4.15–4.68 (3); >5.00 (4) (Bonaparte, 1991; modified from Nesbitt, 2011: 10), ORDERED. This character is inapplicable in taxa with a hooked premaxilla (Figs. 12, 18 and 19).

See comments in Nesbitt (2011: character 10).

29. Premaxilla, downturned main body: absent, alveolar margin sub-parallel to the main axis of the maxilla (0); slightly, in which the alveolar margin is angled at approximately 20°to the alveolar margin of the maxilla (1); strongly, prenarial process obscured by the postnarial process in lateral view (if the postnarial process is long enough) and postnarial process parallel or posteroventrally oriented with respect to the main axis of the premaxillary body (2) (modified from Dilkes, 1998: 6; Reisz, Laurin & Marjanović, 2010: 10; Ezcurra, Lecuona & Martinelli, 2010: 4; Nesbitt, 2011: 8; Dilkes & Arcucci, 2012: 5, in part; Pritchard et al., 2015: 2),ORDERED (Figs. 16–19).

See comments in Nesbitt (2011: character 8).

30. Premaxilla, angle formed between the alveolar margin and the anterior margin of the premaxillary body in lateral view: acute or right-angled (0); obtuse (1) (New character). This character is inapplicable in taxa with a hooked premaxilla (Figs. 20 and 21).

Figure 21 Left (A, C) and both (B, D) premaxillae of Triassic and Recent saurians in (A, C) lateral and (B, D) ventral views.

(A) Planocephalosaurus robinsonae (NHMUK PV R9955); (B) Proterosuchus goweri (NMQR 880); (C) Garjainia madiba (BP/1/6232L; and (D) Salvator merianae (MACN-He 47992). Numbers indicate character-states scored in the data matrix and the arrows indicate anterior direction. Scale bars equal 0.5 mm in (A), 2 cm in (B), 1 cm in (C), and 2 mm in (D).

The anterior tip of the snout is usually acute or 90°in non-pseudosuchian archosauromorphs. By contrast, the angle formed between the anterior margin of the premaxillary body and the alveolar margin of the premaxilla is obtuse in lateral view in Youngina capensis (BP/1/2871, SAM-PK-K7578), basal lepidosauromorphs (e.g., Planocephalosaurus robinsonae: NHMUK PV R9955; Gephyrosaurus bridensis: Evans, 1980), Protorosaurus speneri (Gottmann-Quesada & Sander, 2009), the early rhynchosaur Mesosuchus browni (Dilkes, 1998: Fig. 7A), and ornithosuchid and suchian archosaurs (e.g., Riojasuchus tenuisceps: PVL 3827; Gracilisuchus stipanicicorum: MCZ 4117; Batrachotomus kupferzellensis: SMNS 52970).

31. Premaxilla, longitudinal groove placed approximately at mid-height and extending along most of the length of the lateral surface of the main body of the bone: absent (0); present (1) (New character) (Fig. 21).

The erythrosuchid species Garjainia prima and Garjainia madiba share the presence of a longitudinal and well defined groove that extends along most of the anteroposterior length of the lateral surface of the premaxillary body (PIN 2394/5; BP/1/6232L). This groove is placed approximately at mid-height in the premaxillary body. Some archosauromorphs possess an anteriorly restricted and anteroventrally-to-posterodorsally oriented groove (“Chasmatosaurus” yuani: IVPP V36315) or a more posteriorly restricted groove that is confluent with the posterior margin of the premaxillary body (e.g., Azendohsaurus madagaskarensis: UA 8-7-98-284; Erythrosuchus africanus: BP/1/4526). However, due to the topological differences between the latter conditions and that of Garjainia, it is considered here that this character-state is restricted to this erythrosuchid genus.

32. Premaxilla, narial fossa: absent or shallow (0); expanded in the anteroventral corner of the naris (1) (Sereno, 1999; Nesbitt, 2011: 9; cf. Dilkes & Arcucci, 2012: 7). This character is inapplicable if the premaxilla does not participate in the border of the external naris (Figs. 12, 17 and 20).

See comments in Nesbitt (2011: character 9). Nesbitt (2011) restricted the presence of an expanded narial fossa to dinosauriforms and the loricatan Batrachotomus kupferzellensis, but this condition is found here to be more extensively distributed among Triassic archosauromorphs, being present in Trilophosaurus buettneri (Spielmann et al., 2008), proterosuchids (e.g., Archosaurus rossicus: PIN 1100/55; Proterosuchus fergusi: RC 846, TM 201; Proterosuchus goweri: NMQR 880), Sarmatosuchus otschevi (PIN 2865/68) and proterochampsids (e.g., Proterochampsa nodosa: MCP 1694; Chanaresuchus bonapartei: PULR 07).

33. Premaxilla, peg on the posterior edge of the premaxillary body: absent (0); present (1) (Pritchard et al., 2015: 6) (Fig. 20).

The posterior margin of the premaxillary body possesses a posteriorly opened notch in lateral view in a few archosauromorph species, including Azendohsaurus madagaskarensis (UA 8-7-98-284), Mesosuchus browni (SAM-PK-6536) and the erythrosuchids Erythrosuchus africanus (Gower, 2003; BP/1/4526), Shansisuchus shansisuchus (Young, 1964) and Shansisuchus kuyeheensis (Gower, 2003). At least in Azendohsaurus madagaskarensis (UA 8-7-98-284) and Erythrosuchus africanus (Gower, 2003; BP/1/4526), this notch opens into an anteroventrally oriented groove restricted to the posterior portion of the premaxillary body. Flynn et al. (2010) proposed that this groove likely carried vessel or nerves transmitted by the anteriorly opening foramen of the maxilla.

34. Premaxilla, prenarial process length: less than the anteroposterior length of the main body of the premaxilla (0); greater than the anteroposterior length of the main body of the premaxilla (1) (Nesbitt & Norell, 2006; Nesbitt, 2011: 1). This character is inapplicable in taxa that lack a prenarial process (Figs. 17 and 21).

See comments in Nesbitt (2011: character 1).

35. Premaxilla, base of the prenarial process: anteroposteriorly shallow (0); anteroposteriorly deep (1) (New character). This character is inapplicable if the premaxilla does not participate in the border of the external naris (Figs. 12, 17, 20 and 21).

The base of the prenarial process (=anterodorsal process) of the premaxilla is anteroposteriorly deep in lateral view in the early archosauriforms Archosaurus rossicus (PIN 1100/55), Proterosuchus fergusi (RC 846, TM 201), Proterosuchus goweri (NMQR 880) and “Chasmatosaurus” yuani (IVPP V4067, V36315). In these proterosuchid species, the depth of the prenarial process of the premaxilla immediately anterior to the external naris exceeds half of the length of the narial opening. A similarly deep base of the prenarial process of the premaxilla is present in Youngina capensis (BP/1/2871) and the pterosaur Dimorphodon macronyx (NHMUK PV R41212-13). By contrast, in other archosauromorphs, including Sarmatosuchus otschevi (PIN 2865/68-9), this process is anteroposteriorly shallower and mantains a more constant depth along its proximal half.

36. Premaxilla, postnarial process: absent (0); short, ends well anterior to the posterior margin of the external naris (1); well-developed, forms most of the border of the external naris or excludes the maxilla from participation in the external naris (2) (Laurin, 1991: F1; Reisz, Laurin & Marjanović, 2010: 12; Nesbitt, 2011: cf. 2, 5, 24; Ezcurra, Scheyer & Butler, 2014: 12; Pritchard et al., 2015: 4, 5), ORDERED. This character is inapplicable in taxa with non-terminal external nares (Figs. 17 and 19).

See comments in Nesbitt (2011: characters 2, 5 and 24).

37. Premaxilla, postnarial process: wide, plate-like (0); thin (1) (Gauthier, 1986; Nesbitt, 2011: 3). This character is not applicable to taxa that lack a postnarial process (Fig. 20).

See comments in Nesbitt (2011: character 3).

38. Premaxilla, sharp dorsal flange at the base of the postnarial process delimiting the posteroventral border of the external naris: absent (0); present (1) (New character). This character is inapplicable if the premaxilla does not participate in the border of the external naris (Figs. 12 and 20).

The dorsal margin of the postnarial process (=posteodorsal process) possesses a sharp, dorsally projected flange in Archosaurus rossicus (PIN 1100/55), Proterosuchus fergusi (RC 846, TM 201), Proterosuchus goweri (NMQR 880), “Chasmatosaurus” yuani (IVPP V4067, V36315), Sarmatosuchus otschevi (PIN 2865/68-9), “Chasmatosuchus” vjushkovi (PIN 2394/4) and some specimens of Garjainia madiba (NMQR 3257). This flange forms a laterally offset posterolateral border of the external naris in dorsal view. By contrast, the erythrosuchids Erythrosuchus africanus (BP/1/5207) and Shansisuchus shansisuchus (IVPP V2505) lack this sharp flange at the base of the postnarial process of the premaxilla.

Figure 22 Left (A, D) and right (B, C) maxillae of Triassic saurians in (A, D) lateral and (B, C) medial views.

(A) Planocephalosaurus robinsonae (NHMUK PV R9954); (B) Batrachotomus kupferzellensis (SMNS 52970); (C) Azendohsaurus madagaskarensis (UA 8-29-97-160); and (D) Chalishevia cothurnata (PIN 4366/1, in articulation with the partial left nasal). Numbers indicate character-states scored in the data matrix and the arrows indicate anterior direction. Scale bars equal 1 mm in (A) and 2 cm in (B–D).

39. Premaxilla, postnarial process: fits between the nasal and the maxilla or lies on the anterodorsal surface of the maxilla (0); overlaps the anterodorsal surface of the nasal (1); fits into slot of the nasal (2) (Parrish, 1993; modified from Nesbitt, 2011: 4). This character is not applicable to taxa that lack a postnarial process or it does not extend behind the external naris (Figs. 17 and 23).

Figure 23 Partial skull roofs (A–C, E) and pair of frontals (D, F) of Triassic saurians in (A) dorsal, (B) posterior, and (C–F) ventral views.

(A–C) Erythrosuchus africanus (NMQR 1473); (D) Planocephalosaurus robinsonae (NHMUK PV R9975); (E) Tasmaniosaurus triassicus (UTGD 54655); and (F) Macrocnemus bassanii (PIMUZ T2472). Numbers indicate character-states scored in the data matrix and the arrows indicate anterior direction. Scale bars equal 5 cm in (A–C), 1 mm in (D), 1 cm in (E), and 2 mm in (F).

See comments in Nesbitt (2011: character 4).

40. Premaxilla, contact with prefrontal: absent (0); present, marginal (1); present, extensive (2) (Dilkes, 1998: 7; modified from Ezcurra, Scheyer & Butler, 2014: 114; Pritchard et al., 2015: 4), ORDERED (Fig. 17).

The presence of a contact between the prefrontal and premaxilla is present in rhynchosaurs (Dilkes, 1998) and is a result of a well anteriorly extended prefrontal on an anteroposteriorly short antorbital region. The prefrontal contacts the distal tip of the postnarial process of the premaxilla. In particular, the contact between these bones is only marginal in Mesosuchus browni (Dilkes, 1998) and Howesia browni (Dilkes, 1995), but it is considerably more extensive in the rhynchosaurids Rhynchosaurus articeps (NHMUK PV R1236) and Bentonyx sidensis (BRSUG 21200). A premaxilla-prefrontal contact is also present in the ornithischian dinosaur Heterodontosaurus tucki, but, contrasting with rhynchosaurs, this contact is a result of a posteriorly hypertrophied postnarial process of the premaxilla that terminates close to the anterior border of the orbit and also articulates with the lacrimal (Norman et al., 2011; Sereno, 2012).

41. Premaxilla, palatal process on the medial surface: absent (0); present (1) (Nesbitt et al., 2015: 247) (Figs. 12, 20 and 21).

See comments in Nesbitt et al. (2015): character 247).

42. Premaxilla, number of tooth positions: 10 or more (0); 5 or more (1); 4 (2); 3 (3); 2 (4); 1 or edentulous (5) (Laurin, 1991: G1; Reisz & Dilkes, 2003: 41; Müller, 2004: 152; Reisz, Laurin & Marjanović, 2010: 8; Ezcurra, Lecuona & Martinelli, 2010: 86; Nesbitt, 2011: 6; Ezcurra, Scheyer & Butler, 2014: 8; Pritchard et al., 2015: 88, in part), ORDERED (Figs. 16 and 17).

See comments in Nesbitt (2011: character 6).

43. Premaxilla, orientation of the tooth series or the occlusal surface of premaxilla in ventral view: approximately parasagittal (0); strongly transverse and anterior teeth covering each other in lateral view (1) (New character). This character is inapplicable in taxa without maxillary teeth (Fig. 21).

The premaxilla of some early diapsids (e.g., Petrolacosaurus kansensis: Reisz, 1981; Planocephalosaurus robinsonae: NHMUK PV R9955; Gephyrosaurus bridensis: Evans, 1980; Amotosaurus rotfeldensis: SMNS unnumbered) is anteroposteriorly short and transversely broad, and, as a result, the premaxillary tooth series is mainly transversely oriented in occlusal view. A similar condition occurs in tyrannosaurid theropods (Brochu, 2003). By contrast, the premaxillary tooth series of the archosauromorphs sampled here (with the exception of Amotosaurus rotfeldensis) is mainly parasagittally oriented in occlusal view.

44. Premaxilla, lateroventrally opening anterior alveoli in mature individuals: absent (0); present (1) (New character). This character is inapplicable in taxa without maxillary teeth (Fig. 21).

The premaxillary teeth of almost all the diapsids sampled here are directly ventrally oriented to occlude with the anterior teeth of the lower jaw. However, in adult (or large-sized) specimens of Archosaurus rossicus (PIN 1100/55), Proterosuchus fergusi (RC 846), Proterosuchus goweri (NMQR 880) and “Chasmatosaurus” yuani (IVPP V4067, V36315) the premaxillary teeth are lateroventrally oriented. This condition is not present in smaller, supposedly juvenile specimens of Proterosuchus fergusi (Ezcurra & Butler, 2015b) and it seems to appear later in ontogeny.

45. Septomaxilla: present (0); absent (1) (Dilkes, 1998: 14; Müller, 2004: 87; Reisz, Laurin & Marjanović, 2010: 18) (Fig. 16).

The septomaxilla of saurians is a paired, plate-like bone that forms the floor of the narial chamber and is usually difficult to identify because it is covered by the premaxillae, maxillae and nasals. The septomaxilla curves dorsally towards the midline, where they form nearly vertically oriented sheets. The septomaxilla is present in non-archosauriform diapsids (e.g., Petrolacosaurus kansensis: Reisz, 1981; Youngina capensis: SAM-PK-K7578; Trilophosaurus buettneri: Gregory, 1945; Prolacerta broomi: Modesto & Sues, 2004) and the early archosauriforms Proterosuchus fergusi (CT data of BSPG 1934 VIII 514) and Proterosuchus goweri (NMQR 880). There is no current evidence for the presence of a septomaxilla in any other archosauriform Senter, 2002, and the neomorphic bone present in phytosaurs that is sometimes called septomaxilla is not homologous (Nesbitt, 2011).

46. Maxilla-nasal, maxillo-nasal tuberosity, delimiting anteriorly the antorbital fossa if present: absent (0); present (1) (New character) (Figs. 13, 19 and 22).

The ascending process of the maxilla of the erythrosuchids Garjainia prima (PIN 2394/5-1, PIN 951/32), Erythrosuchus africanus (BP/1/2529, 5207), Shansisuchus shansisuchus (IVPP V2501, V2504, V2505) and Chalishevia cothurnata (PIN 4366/1) possesses a strongly anteroposteriorly convex lateral surface, which forms a vertical tuberosity (also extending onto the nasal and hereafter referred as the maxillo-nasal tuberosity) that defines the anterior margin of the antorbital fossa. As a result, the ascending process has a pillar-like shape in these taxa. A similar condition is present in the suchian Batrachotomus kupferzellensis (SMNS 52970). By contrast, in other archosauromorphs the ascending process of the maxilla is a plate-like structure with a planar or slightly convex external surface (e.g., Prolacerta broomi: BP/1/471; Proterosuchus fergusi: BP/1/4016, BSPG 1934 VIII 514; Fugusuchus hejiapanensis: Cheng, 1980; Euparkeria capensis: SAM-PK-5867).

47. Maxilla-jugal, anguli oris crest: absent (0); present (1) (Benton, 1984b; Butler et al., 2015: 13; Pritchard et al., 2015: 30) (Fig. 16).

The ventral margin of the anterior process of the jugal is distinctly laterally offset and anteromedially-to-posterolaterally oriented from the lateral surface of the maxilla in rhynchosaurids (Benton, 1984b). This offset ventral margin of the jugal forms a distinct shelf that is termed the anguli oris crest (Benton, 1984b). Butler et al. (2015) proposed that a distinct ridge that extends along the maxilla from its contact with the ventral margin of the jugal and terminates anteriorly below the anteroventral corner of the orbit in Eohyosaurus wolvaardti is homologous to the anguli oris crest of rhynchosaurids. This crest is not present in other Permo-Triassic archosauromorphs.

48. Maxilla-jugal, anterior extension of the anguli oris crest: restricted to the main body of the jugal (0); extending onto the maxilla, but not the anterior process of the jugal (1) (Benton, 1984b; Butler et al., 2015: 14). This character is inapplicable in taxa without an anguli oris crest (Fig. 16).

In Eohyosaurus wolvaardti (SAM-PK-K10159) and Rhynchosaurus articeps (NHMUK PV R1236, SHYMS 1) the anguli oris crest extends anteriorly on the lateral surface of the maxilla from its suture with the anterior process of the jugal. By contrast, the anguli oris crest is restricted to the ventral margin of the anterior process and main body of the jugal in Bentonyx sidensis, Stenaulorhynchus stockleyi, Fodonyx spenceri, and hyperodapedontines (Butler et al., 2015).

49. Maxilla, anterior extent: posterior to the anterior extent of the nasal (0); anterior to the nasal (1) (Sereno, 1991; Nesbitt, 2011: 19) (Fig. 16 of Nesbitt, 2011).

See comments in Nesbitt (2011: character 19).

50. Maxilla, length of the portion of the bone anterior to the antorbital fenestra versus the total length of the bone: 0.12–0.22 (0); 0.29–0.60 (1); 0.64–0.76 (2) (Clark, Sues & Berman, 2001; modified from Nesbitt, 2011: 14), ORDERED. This character is not applicable in taxa that lack an antorbital fenestra (Fig. 18).

See comments in Nesbitt (2011: character 14).

51. Maxilla, posterior border of the subnarial foramen extending posteriorly as a groove on the lateral surface of the anterior process: absent (0); present (1) (New character). This character is not applicable in taxa that lack a subnarial foramen bordered by the maxilla (Figs. 17 and 22).

The lateral surface of the anterior process of most archosauromorphs is mostly flat. In “Chasmatosaurus” yuani (IVPP V4067, V36315) and some erythrosuchids (e.g., Garjainia prima: PIN 2394/5, Shansisuchus shansisuchus: Wang et al., 2013; Chalishevia cothurnata: PIN 2867/7) the lateral surface of the anterior process of the maxilla possesses a mainly longitudinal groove that merges with a notch on the margin of the bone. This notch is considered homologous to the notched anterior margin of the maxilla that contributes to a fully enclosed subnarial foramen, by both premaxilla and maxilla, in other archosaurormophs.

52. Maxilla, anterior maxillary foramen: absent (0); present (1) (Dilkes, 1998: 17; Müller, 2004: 88; Modesto & Sues, 2004; Nesbitt, 2011: 31; Ezcurra, Scheyer & Butler, 2014: 120; Nesbitt et al., 2015: 203) (Fig. 17).

See comments in Nesbitt (2011: character 31).

53. Maxilla, neurovascular foramina on the lateral surface of the anterior and horizontal processes: laterally or lateroventrally facing (0); lateroventrally facing and extending ventrally as deep, well-defined grooves (1) (New character). This character is inapplicable in taxa lacking neurovascular foramina on the lateral surface of the maxilla (Figs. 19 and 22).

The lateral surface of the maxilla of the sampled diapsids is perforated by multiple laterally or lateroventrally opening foramina aligned in a more or less longitudinal row that is parallel to the alveolar margin. These foramina presumably transmitted the cutaneous branch of the maxillary artery and nerve (e.g., Gephyrosaurus bridensis: Evans, 1980). In particular, in erythrosuchids (e.g., Guchengosuchus shiguaiensis: IVPP V8808; Garjainia prima: PIN 2394/5; Erythrosuchus africanus: BP/1/5207; Chalishevia cothurnata: PIN 2867/7), the enigmatic Youngosuchus sinensis (IVPP V3239) and Asperoris mnyama (NHMUK PV R36615), and some predatory archosaurs (e.g., Batrachotomus kupferzellensis: SMNS 52970; Herrerasaurus ischigualastensis: PVSJ 407) these foramina extend ventrally as relatively long, deep and well defined grooves. These long grooves are absent in most diapsids.

54. Maxilla, antorbital fossa on the lateral surface of the bone: absent or not exposed in lateral view (0); present on the ascending process of the maxilla, but not along the horizontal process of the maxilla (1); present on the horizontal process of the maxilla, but not reaching the posteroventral corner of the fenestra (2); present on the horizontal process of the maxilla, reaching the posteroventral corner of the opening (3) (modified from Benton, 2004; Dilkes & Sues, 2009: 3; Nesbitt, 2011: 137, in part; Dilkes & Arcucci, 2012: 3 and 4; Ezcurra, Scheyer & Butler, 2014: 119; Nesbitt et al., 2015: 212), ORDERED. This character is not applicable in taxa that lack an antorbital fenestra (Figs. 13, 17–19 and 22).

See comments in Nesbitt (2011: character 137).

55. Maxilla, anteroposterior length of the antorbital fossa anterior to the antorbital fenestra versus length of the antorbital fenestra: 0.09–0.23 (0); 0.28–0.43 (1); 0.90–0.94 (2); >2.00 (3) (modified from Sereno et al., 1994), ORDERED. This character is inapplicable in taxa that lack an antorbital fossa or where the fossa is not extended anterior to the antorbital fenestra (Figs. 18 and 19).

This character describes the degree of anteroposterior extension of the antorbital fossa immediately anterior to the antorbital opening.

56. Maxilla, secondary antorbital fossa anteriorly to the antorbital fossa and adjacent to the dorsal margin of the anterior process: absent (0); present (1) (New character). This character is not applicable in taxa that lack a secondary antorbital fenestra (Fig. 22).

Among the erythrosuchids that possess a secondary antorbital fenestra, Shansisuchus shansisuchus (IVPP V2501, V2504, V2505; Young, 1964; Wang et al., 2013) and Chalishevia cothurnata (PIN 2867/7) share the presence of a secondary antorbital fossa placed ventral to the secondary antorbital opening. This fossa is delimited posteriorly by the maxillo-nasal tuberosity and ventrally by a posterodorsally-to-anteroventrally oriented rim. The erythrosuchid Guchengosuchus shiguaiensis possesses a secondary antorbital fenestra but lacks a secondary antorbital fossa (IVPP V8808).

57. Maxilla, ascending process: absent (0); present (1) (Reisz & Dilkes, 2003: 5; Reisz, Laurin & Marjanović, 2010: 13; modified from Ezcurra, Scheyer & Butler, 2014: 13).

The maxilla of saurians possesses a dorsally projected flange that is differentiated from a posterior alveolar portion of the bone by a mainly dorsoventrally oriented margin. By contrast, in Petrolacosaurus kansensis (Reisz, 1981) and Youngina capensis (Gow, 1975) the maxilla possesses a broad, anteroposteriorly convex dorsal margin that is not distinctly differentiated from the posterior alveolar portion of the bone.

58. Maxilla, ascending process shape: simply tapers to a point dorsally (0); the dorsal apex of the maxilla is a separate, distinct process that has a posteriorly concave margin (1); sub-vertical anterior margin of the base of the process (2) (Reisz & Dilkes, 2003: 5; Reisz, Laurin & Marjanović, 2010: 13; modified from Ezcurra, Scheyer & Butler, 2014: 13, and Nesbitt et al., 2015: 202) (Fig. 22).

See comments in Nesbitt et al. (2015: character 202).

59. Maxilla, anterodorsal margin at the base of the ascending process: convex or straight (0); concave (1) (Langer & Benton, 2006; Nesbitt, 2011: 25). This character is not applicable in taxa that lack an ascending process or possess a secondary antorbital fenestra (Fig. 22, Figs. 15, 17 and 19 of Nesbitt, 2011).

See comments in Nesbitt (2011: character 25).

60. Maxilla, ascending process remains the same width for its length: absent (0); present (1) (Nesbitt, 2011: 29). This character is not applicable in taxa that lack an ascending process or an antorbital fenestra (Fig. 22, Figs. 15, 17, 19 of Nesbitt, 2011).

See comments in Nesbitt (2011: character 29).

61. Maxilla, contact with prefrontal: absent (0); present (1) (Reisz & Dilkes, 2003: 6; Müller, 2004: 179; Reisz, Laurin & Marjanović, 2010: 14; Dilkes & Arcucci, 2012: 9 and 10; Ezcurra, Scheyer & Butler, 2014: 14; Pritchard et al., 2015: cf. 12) (Fig. 17).

The anterior portion of the prefrontal contacts the ascending process of the maxilla because of the small size of the lacrimal in basal lepidosauromorphs (e.g., Gephyrosaurus bridensis: Evans, 1980), choristoderans (e.g., Cteniogenys: Evans, 1990), tanystropheids (e.g., Tanystropheus longobardicus: Wild, 1973), Trilophosaurus buettneri (Spielmann et al., 2008), and rhynchosaurs (e.g., Mesosuchus browni: Dilkes, 1998; Rhynchosaurus articeps: Benton, 1990). This condition is also present in Vancleavea campi, in which the lacrimal is absent and, as a result, the prefrontal extensively contacts the maxilla (Nesbitt et al., 2009). In proterochampsids, the prefrontal also contacts the maxilla, but it is mainly a result of the strong anterior development of the former bone rather than a reduction of the lacrimal (Romer, 1971a; Dilkes & Arcucci, 2012; Trotteyn, Arcucci & Raugust, 2013). The condition in Vancleavea campi and proterochampsids were scored as the same due to the similar arrangement of the bone contacts. In the other diapsids sampled here, a maxilla-prefontal contact is prevented by a lacrimal-nasal contact.

62. Maxilla, ventral margin of the antorbital fossa or fenestra (if the antorbital fossa is absent from the horizontal process of the maxilla) in the horizontal process: mainly sub-parallel to the alveolar margin of the bone (0); diagonal, anteroventrally-to-posterodorsally oriented in an angle close to 45°(1) (New character). This character is inapplicable in taxa that lack an antorbital fenestra or fossa (Figs. 17 and 22).

The ventral margin of the antorbital fenestra of Garjainia prima is straight and anteroventrally-to-posterodorsally oriented at an angle of approximately 45°with respect to the longitudinal axis of the skull (PIN 2394/5). This condition results in a horizontal process of the maxilla that abruptly increases in height posteriorly. In Shansisuchus shansisuchus (IVPP V2501, V2504, V2505) and Chalishevia cothurnata (PIN 2867/7), the ventral margin of the antorbital fenestra is gently concave and mainly anteroposteriorly oriented in lateral view, but the ventral margin of the antorbital fossa closely resembles the step anteroventrally-to-posterodorsally orientation present in the ventral margin of the antorbital fenestra of Garjainia prima. Since Garjainia prima lacks an antorbital fossa, it is hypothetized here that the diagonal ventral border of the antorbital fenestra of Garjainia prima and the antorbital fossa of Shansisuchus shansisuchus and Chalishevia cothurnata can be considered primary homologous conditions. No other archosauriform sampled here possesses such a steep angle of the ventral margin of the antorbital fossa or fenestra.

63. Maxilla, shape of the posterior portion of the bone ventral to the antorbital fenestra: tapers posteriorly (0); has a similar dorsoventral depth as the anterior portion ventral to the antorbital fenestra (1); expands dorsoventrally towards the distal end of the horizontal process with a concave ventral margin of the antorbital fenestra (2); expands dorsoventrally towards the distal end of the horizontal process with a straight ventral margin of the antorbital fenestra (3) (modified from Nesbitt, 2011: 27). This character is inapplicable in taxa that lack an antorbital fenestra (Figs. 18, 19 and 22).

See comments in Nesbitt (2011: character 27).

64. Maxilla, posterior end of the horizontal process distinctly ventrally deflected from the main axis of the alveolar margin: absent (0); present (1) (New character) (Figs. 17 and 22).

The posterior end of the maxilla, immediately posterior to the alveolar portion of the bone, is posteroventrally deflected in the allokotosaurians Azendohsaurus madagaskarensis (Flynn et al., 2010) and Trilophosaurus buettneri (Spielmann et al., 2008). In other diapsids, the ventral margin of the posterior end of the maxilla is approximately aligned with the alveolar margin of the bone.

65. Maxilla, triangular dorsal process with clear dorsal apex formed by discrete expansion (=posterodorsal process sensu Butler et al., 2014a) of the posterior end of the horizontal process in lateral view: absent (0); present (1) (modified from Butler et al., 2014a: 413) (Fig. 18).

Among the archosauromorphs sampled here, the gracilisuchids Gracilisuchus stipanicicorum and Turfanosuchus dabanensis (Butler et al., 2014a), the aetosaur Aetosauroides scagliai (Desojo & Ezcurra, 2011), the loricatan Prestosuchus chiniquensis (UFRGS-PV-0156-T), and the enigmatic Youngosuchus sinensis (IVPP V3239) possess a pronounced “posterodorsal process” at the posterior end of the horizontal process, posterior to the antorbital fenestra and fossa. This posterodorsal process forms a dorsoventrally deep, symmetrical, triangular projection with anterodorsal and posterodorsal margins that form steep angles to the horizontal (Butler et al., 2014a). By contrast, in most archosauriforms the maxilla either tapers or maintains a nearly constant depth towards its posterior end.

66. Maxilla, palatal process on the anteromedial surface of the bone: absent (0); present and both counterparts do not meet at the midline (1); present and both counterparts meet at the midline (2) (Parrish, 1993; Gower & Sennikov, 1997; Ezcurra, Lecuona & Martinelli, 2010: 101; Nesbitt, 2011: 32; Dilkes & Arcucci, 2012: 13, in part; Nesbitt et al., 2015: 204), ORDERED (Figs. 13 and 22, Fig. 15 of Nesbitt, 2011).

The description and scorings of this character closely agree with those of Nesbitt (2011: character 32), who scored the presence of a palatal process of the maxilla in erythrosuchids and more crownward archosauriforms. However, contrasting with the scoring of this author, the rhynchosaur Mesosuchus browni (SAM-PK-6536) and the proterosuchid archosauriforms Proterosuchus fergusi (unpublished CT-data of BSPG 1934 VIII 514), Proterosuchus goweri (NMQR 880) and “Chasmatosaurus” yuani (IVPP V36315) possess a palatal process that is developed as an anteriorly projected medial shelf on the anterior tip of the maxilla.

67. Maxilla, position of the palatal process: adjacent to the bases of the interdental plates (0); distinctly dorsal to the bases of the interdental plates (1) (New character). This character is inapplicable in taxa lacking a palatal process on the maxilla (Figs. 13 and 22).

The palatal process of the maxilla is placed either adjacent to the base of the interdental plates (e.g., Proterosuchus goweri: NMQR 880, “Chasmatosaurus” yuani: IVPP V36315; Kalisuchus rewanensis: QM F8998; Guchengosuchus shiguaiensis: IVPP V8808; Garjainia prima: PIN 2394/5; Euparkeria capensis: SAM-PK-6050; Lewisuchus admixtus: PULR 01) or distinctly dorsal to the alveolar margin of the bone and adjacent to the anterodorsal margin of the anterior process (e.g., Erythrosuchus africanus: BP/1/4680, SAM-PK-K1098; Asperoris mnyama: NHMUK PV R36615; Yarasuchus deccanensis: ISI R334-2; Batrachotomus kupferzellensis: SMNS 52970; Herrerasaurus ischigualastensis: PVSJ 53).

68. Maxilla, alveolar margin in lateral view: concave, straight or gently convex (0); distinctly convex (1); sigmoid, anteriorly concave and posteriorly convex (2); sigmoid, anteriorly convex, starting close to mid-length, and posteriorly concave (3) (Dilkes, 1998: 16; modified from Ezcurra, Scheyer & Butler, 2014: 121, and Pritchard et al., 2015: 7) (Figs. 16 and 19).

The alveolar margin of the maxilla is concave, straight or gently convex along its extension in lateral view in most diapsids. In particular, the ventral margin of the maxilla of Howesia browni and rhynchosaurids is strongly convex and, as a result, the anterior portion of the maxilla is placed distinctly dorsal to the level of its posterior end (Benton, 1990; Dilkes, 1995; Dilkes, 1998). In a disparate array of archosauromorphs, the ventral margin of the maxilla is sigmoid in lateral view, being either anteriorly concave and posteriorly convex (e.g., Prolacertoides jimusarensis: IVPP V3233; Proterosuchus goweri: NMQR 880; Chalishevia cothurnata: PIN 2867/7, phytosaurs: ISI R42, UCMP 27200) or anteriorly convex and posteriorly concave (e.g., Tanystropheus longobardicus: Wild, 1973; Erythrosuchus africanus: BP/1/5207; Yarasuchus deccanensis: ISI R334-2; Herrerasaurus ischigualastensis: PVSJ 53, 407).

69. Maxilla, edentulous anterior portion of the ventral margin of the bone: absent (0); present (1) (New character) (Figs. 13 and 17).

The anterior end of the ventral margin of the maxilla of some archosauromorphs possesses an edentulous gap equivalent to at least one or two tooth positions (e.g., Trilophosaurus buettneri: (Spielmann et al., 2008); Mesosuchus browni: SAM-PK-6536; Proterosuchus goweri: NMQR 880; “Chasmatosaurus” yuani: IVPP V4067, V36315; Garjainia prima: PIN 2394/5; Erythrosuchus africanus: BP/1/5207; Vancleavea campi: Nesbitt et al., 2009; Gualosuchus reigi: PULR 05; Heterodontosaurus tucki: SAM-PK-K337). By contrast, the maxillary tooth row in most archosauromorphs reaches the anterior tip of the ventral margin of the maxilla and usually is continuous with the premaxillary tooth series.

70. Maxilla, alveolar margin on the anterior third of the bone (anterior to the level of the anterior border of the antorbital fenestra if present): approximately aligned to the posterior half of the alveolar margin (0); abruptly upturned (1) (New character) (Figs. 13, 17 and 19).

The anterior portion of the alveolar margin of the maxilla is abruptly upturned with respect to the rest of the alveolar margin of the bone, contrasting with the more gradual upturning present in convex or sigmoid alveolar margins, in “Chasmatosaurus” ultimus (IVPP V2301), Youngosuchus sinensis (IVPP V3239), erythrosuchids (e.g., Garjainia prima: PIN 2394/5; Erythrosuchus africanus: BP/1/5207), and some suchians (e.g., Gracilisuchus stipanicicorum: MCZ 4117; Turfanosuchus dabanensis: IVPP V3237; Batrachotomus kupferzellensis: SMNS 52970). By contrast, in most archosauromorphs, the anterior portion of the alveolar margin of the maxilla is rather straight in lateral view or retains the degree of convexity present in the more posterior portion of the bone (e.g., Proterosuchus goweri: NMQR 880).

71. Maxilla, posterior extension in mature individuals: level with or posterior to posterior orbital border (0); anterior to posterior orbital border but posterior to anterior orbital border (1); level with or anterior to anterior orbital border (2) (DeBraga & Rieppel, 1997; Müller, 2004: 127; modified from Ezcurra, Scheyer & Butler, 2014: 122), ORDERED (Figs. 14 and 19).

The posterior tip of the maxilla of the vast majority of diapsids sampled here is placed anterior to the level of the posterior orbital border, but posterior to the anterior orbital border. By contrast, the maxilla extends posteriorly to the level of the posterior orbital border in proterosuchids (e.g., Proterosuchus goweri: RC 846; Proterosuchus goweri: NMQR 880; “Chasmatosaurus” yuani: IVPP V4067), Vancleavea campi (Nesbitt et al., 2009) and Doswellia kaltenbachi: USNM 214823). In other taxa, the maxilla is restricted to the snout, as occurs in Proterochampsa barrionuevoi (Dilkes & Arcucci, 2012), Parasuchus hislopi (ISI R42) and Dimorphodon macronyx (NHMUK PV R41212).

72. Maxilla, tooth plate: absent (0); present (1) (Dilkes, 1998: 60; Ezcurra, Scheyer & Butler, 2014: 124) (Fig. 14).

This character is present in all rhynchosaurs to the exclusion of Mesosuchus browni (Dilkes, 1998). See Benton (1984b) for a description of the maxillary tooth plate.

73. Maxilla, number of tooth rows: single row (0); multiple rows (1) (Dilkes, 1998: 61; Ezcurra, Scheyer & Butler, 2014: 125; Pritchard et al., 2015: 92) (Fig. 14).

Rhynchosaurs are characterized by the presence of multiple, longitudinal tooth rows (Chatterjee, 1978; Benton, 1984b; Dilkes, 1998), a condition that resembles that of the Palaeozoic captorhinid sauropsids (De Ricqlès & Bolt, 1983) but clearly differs from the morphology present in other archosauromorphs. See a discussion of this character in Benton (1984b) and Dilkes (1998).

74. Maxilla, location of teeth: only on occlusal surface (0); on occlusal and lingual surfaces (1) (Dilkes, 1998: 63; Ezcurra, Scheyer & Butler, 2014: 126) (Fig. 14).

Most rhynchosaur species possess teeth on both occlusal and ligual surfaces of the maxilla (i.e., Howesia browni, Eohyosaurus wolvaardti, non-Hyperodapedon rhynchosaurids, and some species of Hyperodapedon; Langer & Schultz, 2000; Hone & Benton, 2008; Mukherjee & Ray, 2014; Butler et al., 2015), which are involved in a Zahnreihen model of tooth growth and replacement as described by Edmund (1960). See the description and discussion of this character in Benton (1984b) and Dilkes (1998).

75. Maxilla, number of tooth positions: 8–9 (0); 10–14 (1); 15–22 (2); 23–35 (3); 36–40 (4) (modified from Reisz & Dilkes, 2003: 28; Reisz, Laurin & Marjanović, 2010: 15; Ezcurra, Scheyer & Butler, 2014: 15), ORDERED. This character is inapplicable in taxa with multiple tooth rows in the maxilla.

76. Nasal, total length versus total length of the frontal: 0.68–0.79 (0); 0.92–2.07 (1); 2.26–3.09 (2) (Dilkes, 1998; Rieppel, Mazin & Tchernov, 1999 in part; Reisz & Dilkes, 2003: 50 in part; Müller, 2004: 4, 8; modified from Reisz, Laurin & Marjanović, 2010: 6, 22; Ezcurra, Lecuona & Martinelli, 2010: 9, in part; modified from Dilkes & Arcucci, 2012: 14; modified from Ezcurra, Scheyer & Butler, 2014: 6), ORDERED (Fig. 16).

77. Nasal, exposure (excluding descending process if present): largely dorsal element (0); nearly vertical contribution to the snout (1) (reworded from Reisz & Dilkes, 2003: 12; Reisz, Laurin & Marjanović, 2010: 5; Ezcurra, Scheyer & Butler, 2014: 5).

Trilophosaurus buettneri possesses a relatively tall skull relative to its length, in which the nasal extends considerably ventrally and widely contributes to the lateral surface of the snout between the external naris and the remainder of the snout (Spielmann et al., 2008). In other diapsids sampled here, the nasal is largely restricted to the skull roof.

78. Nasal, shape of anterior margin at midline: strongly convex with anterior process (0); transverse with little convexity (1) (Dilkes, 1998: 13; Ezcurra, Scheyer & Butler, 2014: 127) (Fig. 16).

Howesia browni and rhynchosaurids lack an anterodorsal process, contributing to the internarial bar, contrasting with the condition in the vast majority of diapsids (Dilkes, 1995; Dilkes, 1998).

79. Nasal, anterior portion in lateral view: below or at the same level as skull roof (0); elevated above skull roof, giving the skull a “Roman nose” appearance (1) (Gower, 1999; Brusatte et al., 2010: 25).

The anterior portion of the nasal is dorsally elevated above the rest of the skull roof and results in a broadly convex dorsal margin of the snout approximately above the level of the external naris in Youngosuchus sinensis (IVPP V3239), Riojasuchus tenuisceps (PVL 3827), Turfanosuchus debanensis (IVPP V3237), Nicrosaurus kappfi (NHMUK PV R42743), and rauisuchians (Gower, 1999; Brusatte et al., 2010).

80. Nasal, dorsal surface around posterior margin of external naris: smooth or sculpturing of ridges and grooves present (0); depression around entire posterior margin that lacks sculpturing (1) (Dilkes & Arcucci, 2012: 8) (Fig. 16).

See comments in Dilkes & Arcucci (2012: 28).

81. Nasal, descending process, which results from the articulation of the postnasal process of the premaxilla on the anterodorsal surface of the nasal and has an extensive contact with the ascending process of the maxilla: anteroposteriorly narrow (0); anteroposteriorly very broad, being considerably broader than the ascending process of the maxilla (1) (New character). This character is scored inapplicable in taxa that lack a descending process (Figs. 17 and 23).

In several non-proterosuchid archosauriforms, a long postnasal process of the premaxilla extensively overlaps the anterodorsal surface of the nasal. This contact results in a bifurcation of the anterodorsal surface of the nasal into an anterior process that forms the posteroventral border of the external naris and a descending process that separates the postnasal process of the premaxilla from the ascending process of the maxilla (e.g., Guchengosuchus shiguaiensis: IVPP V8808; Erythrosuchus africanus: BP/1/5207, Asperoris mnyama: NHMUK PV R36615; Euparkeria capensis: SAM-PK-5867; Riojasuchus tenuisceps: PVL 3827; Turfanosuchus debanensis: IVPP V3237; Gracilisuchus stipanicicorum: MCZ 4117; Dimorphodon macronyx: NHMUK PV R41212). The descending process of the nasal is anteroposteriorly narrow in most species, but in Garjainia prima (PIN 2394/5), Shansisuchus shansisuchus (Young, 1964; Wang et al., 2013) and Chalishevia cothurnata (PIN 2867/7) it is anteroposteriorly broader than the ascending process of the maxilla.

82. Nasal, dorsolateral margin of the anterior portion: smoothly rounded (0); distinct longitudinal ridge on the lateral edge (1) (Nesbitt, 2011: 35) (Figs. 19 and 21 of Nesbitt, 2011).

See comments in Nesbitt (2011: character 35).

83. Nasal, participation in the dorsal border of the antorbital fossa: absent (0); present (1) (Sereno et al., 1994; Nesbitt, 2011: 37). This character is inapplicable in taxa that lack an antorbital fossa (Fig. 18).

See comments in Nesbitt (2011: character 37).

84. Lacrimal-postorbital, contact between bones: absent (0); present (1) (New character). This character is inapplicable in taxa that lost the lacrimal.

In several phytosaurs, including Smilosuchus spp. and some specimens of Nicrosaurus kapffi, the lacrimal possesses a posterior process that forms the ventral border of the orbit and participates in an extensive diagonal, anteroventrally-to-posterodorsally oriented suture with the ascending process of the jugal (Camp, 1930; Hungerbühler, 1998). The distal end of this posterior process of the lacrimal contacts the ventral process of the postorbital at the posteroventral border of the orbit.

85. Lacrimal, participation in the posterior border of the external naris: present (0); absent (1) (reworded from Laurin, 1991: B2; Reisz, Laurin & Marjanović, 2010: 24; Ezcurra, Lecuona & Martinelli, 2010: 6, in part; Ezcurra, Scheyer & Butler, 2014: 23; Nesbitt et al., 2015: 11; Pritchard et al., 2015: 11). This character is inapplicable in taxa lacking a lacrimal.

In the early diapsid Petrolacosaurus kansensis the maxilla is restricted to a dorsoventrally low and long bone that does not reach the level of the mid-height of the snout. Conversely, the lacrimal is the most laterally exposed bone of the snout of this species and extends anteriorly to form the posterior border of the external naris (Reisz, 1981). In all the other diapsids sampled in this analysis, the lacrimal does not reach the posterior border of the naris, and this margin is formed by the maxilla and nasal.

86. Lacrimal, exposure on the skull roof in dorsal view: absent or marginal (0); present (1) (Ezcurra, Lecuona & Martinelli, 2010: 166). This character is inapplicable in taxa lacking a lacrimal.

The lacrimal is restricted to the lateral surface of the skull or participates only marginally (exposed as a very narrow splint of bone) in all the non-archosaurian diapsids sampled here. In some archosaurs, including ornithosuchids (PVL 3827), gracilisuchids (IVPP V3237, MCZ 4117), Aetosauroides scagliai (PVL 2059), Silesaurus opolensis (Dzik, 2003) and Herrerasaurus ischigualastensis (PVSJ 407), the lacrimal is broadly exposed in the skull roof, being slightly narrower or subequal in width to the prefrontal.

87. Lacrimal, anterior process forming the entire or almost the entire dorsal border of the antorbital fenestra: absent (0); present (1) (New character). This character is not applicable in taxa that lack an antorbital fenestra or lacrimal (Fig. 17).

The development of a distinct anterior process in the lacrimal is mainly a result of the presence of an antorbital fenestra in archosauriforms. The anterior process of the lacrimal is restricted to the posterodorsal border of the antorbital opening in most species. However, this process is considerably anteroposteriorly elongated and forms the entire or almost the entire dorsal border of the antorbital fenestra in some proterosuchians (e.g., Proterosuchus fergusi: RC 846, SAM-PK-11208; Proterosuchus goweri: NMQR 880; “Chasmatosaurus” yuani: IVPP V4067; Tasmaniosaurus triassicus: UTGD 54655; Fugusuchus hejiapanensis: Cheng, 1980), Euparkeria capensis (SAM-PK-5867), Tropidosuchus romeri (PVL 4606), gracilisuchids (Turfanosuchus dabanensis: IVPP V3237; Gracilisuchus stipanicicorum: MCZ 4117) and Heterodontosaurus tucki (SAM-PK-K337).

88. Lacrimal, antorbital fossa forming a distinct inset margin to the antorbital fenestra on the lateral surface of the bone: absent (0); present, but strongly restricted anteirorly (1); present and occuping almost half or more of the anteroposterior length of the ventral process (2) (Benton, 2004; modified from Ezcurra, Lecuona & Martinelli, 2010: 2; modified from Trotteyn & Ezcurra, 2014: 108). This character is not applicable in taxa that lack an antorbital fenestra or lacrimal (Fig. 18).

The antorbital fossa of the lacrimal is formed by a well-rimmed, concave depression on the lateral surface of the bone and adjacent to the posterodorsal and posterior borders of the antorbital fenestra. This fossa is present in the vast majority of Triassic archosauriforms, with the exception of some proterochampsids (Gualosuchus reigi: PULR 05; Tropidosuchus romeri: PVL 4601) and some phytosaurs (e.g., Nicrosaurus kapffi: NHMUK PV R42743; Smilosuchus gregorii: UCMP 27200). In most archosauriforms, the antorbital fossa is strongly restricted anteriorly on the ventral process of the lacrimal, occupying less than half the anteroposterior length of the process. However, in Youngosuchus sinensis (IVPP V3239), some proterochampsids (Cerritosaurus binsfeldi: UFRGS cast of CA s/n; Chanaresuchus bonapartei: PULR 07; Rhadinosuchus gracilis: BSPG AS XXV 50), some basal phytosaurs (e.g., Parasuchus hislopi: ISI R42; Parasuchus angustifrons: BSPG 1931 X 502), and ornithodirans (e.g., Dimorphodon macronyx: NHMUK PV R41212; Silesaurus opolensis: Dzik & Sulej, 2007; Herrerasaurus ischigualastensis: PVSJ 407; Heterodontosaurus tucki: SAM-PK-K337) the antorbital fossa occupies more than half of the anteroposterior length of the ventral process of the lacrimal, with an outer rim placed closer to the orbital border than to the antorbital margin.

89. Lacrimal, naso-lacrimal duct: completely enclosed by the lacrimal (0); enclosed by the lacrimal and prefrontal (1) (New character) (Figs. 17 and 19).

The foramen or foramina for the exit of the naso-lacrimal duct are placed immediately anterior to the anterior border of the orbit, and they are completely enclosed by the lacrimal in the non-archosauriform diapsids sampled here (e.g., lepidosauromorphs: Evans, 1980; rhynchosaurs: Benton, 1990; Prolacerta broomi: Modesto & Sues, 2004) and in Proterosuchus fergusi (RC 846) and Proterosuchus goweri (NMQR 880). By contrast, the exit of the naso-lacrimal duct is enclosed by both lacrimal and prefrontal in the very few non-proterosuchid archosauriforms where it can be confidently observed, namely Garjainia prima (PIN 2394/5), Euparkeria capensis (Senter, 2003), and Batrachotomus kupferzellensis (Gower, 1999).

90. Lacrimal, naso-lacrimal duct position: opens on the posterolateral edge of the lacrimal (0); opens on the posterior surface of the lacrimal (1) (Reisz & Dilkes, 2003: 19; cf. Müller, 2004: 129; Reisz, Laurin & Marjanović, 2010: 25; Ezcurra, Lecuona & Martinelli, 2010: 8; Dilkes & Arcucci, 2012: 12, in part; modified from Ezcurra, Scheyer & Butler, 2014: 24). This character is inapplicable if the prefrontal encloses part of the naso-lacrimal duct (Fig. 19).

The exit of the naso-lacrimal duct opens on the posterolateral edge of the lacrimal and is broadly visible in lateral view in most diapsids (e.g., Gephyrosaurus bridensis: Evans, 1990; Proterosuchus fergusi: RC 846). Among the taxa sampled here, the foramen or foramina for the exit of the naso-lacrimal duct open posteriorly on the posterior surface of the lacrimal in Cteniogenys Evans, 1990 and rhynchosaurs (e.g., Mesosuchus browni: Dilkes, 1998; Rhynchosaurus articeps: Benton, 1990).

91. Jugal-quadratojugal, ventral margin in lateral view: straight or convex (0); concave, though nowhere dorsal to tooth row (1) (Reisz & Dilkes, 2003: 52; Reisz, Laurin & Marjanović, 2010: 42; Ezcurra, Scheyer & Butler, 2014: 41). Scored as inapplicable in taxa that lack the posterior process of the jugal.

The ventral margin of the postorbital region of the skull is convex or straight in the vast majority of diapsids. However, in Youngina capensis (BP/1/3859), Doswellia kaltenbachi (USNM 214823), phytosaurs (e.g., Parasuchus hislopi: ISI R42; Smilosuchus gregorii: UCMP 27200), and Silesaurus opolensis (Dzik & Sulej, 2007) the ventral margins of the jugal and the anterior process of the quadratojugal are concave and, as a result, extend dorsally to the level of the posterior end of the alveolar margin of the maxilla.

92. Jugal, anterior process shape in lateral view: continuously tapering or subrectangular, being lower than the portion of the maxilla underneath it (0); subrectangular or slightly dorsoventrally expanded, being higher than the portion of the maxilla underneath it (1); with an ascending subprocess excluding the lacrimal from the anteroventral border of the orbit (2) (Gower & Sennikov, 1997; Nesbitt et al., 2009: 12; Ezcurra, Lecuona & Martinelli, 2010: 99; modified from Nesbitt et al., 2015: 205) (Figs. 17 and 19).

This character has previously been discussed by Nesbitt et al. (2009: character 12). Here a third character-state was added to account for the presence of a dorsal subprocess on the anterior process of the jugal, which forms the anteroventral border of the orbit, in Garjainia prima (PIN 2394/5), Erythrosuchus africanus (BP/1/5207), ornithosuchids (Walker, 1964; Bonaparte, 1982; Baczko & Ezcurra, 2013) and Dimorphodon macronyx (NHMUK PV R41212).

93. Jugal, anterior process continuously dorsally curved: absent, straight or curved only at its proximal half (0); present (1) (New character) (Fig. 19).

The anterior process of the jugal is straight or curves dorsally only in its distal half, along the anteroventral border of the orbit, in most diapsids. By contrast, the anterior process of the jugal is continuously dorsally curved, from its base, in lateral view in disparate diapsid taxa sampled here, incuding Youngina capensis (BP/1/2871), Paliguana whitei (AM 3585), Tanystropheus longobardicus (Nosotti, 2007), Rhynchosaurus articeps (NHMUK PV R1236, SHYMS 1), Prolacerta broomi (BP/1/471) and the proterosuchids Proterosuchus fergusi (SAM-PK-11208; RC 846), Proterosuchus alexanderi (NMQR 880) and “Chasmatosaurus” yuani (IVPP V4067).

94. Jugal, ventral border of the orbit: gently concave (0); V-shaped (1) (New character). This character is inapplicable in taxa in which the jugal does not contribute to the border of the orbit (Figs. 19 and 24).

Figure 24 Cranial bones of Triassic saurians in (A–C) lateral, (D) medial, and (E) posterior views.

(A) Right squamosal in articulation with the quadrate of Sarmatosuchus otschevi (PIN 2865/68-3, 4, mirrored); (B) left jugal of Sarmatosuchus otschevi (PIN 2865/68-6); (C) left quadrate of Pamelaria dolichotrachela (ISI R316/1); (D) right palatine of Azendohsaurus madagaskarensis (UA 8-27-98-273); and (E) left quadrate and probably fused quadratojugal of Planocephalosaurus robinsonae (NHMUK PV R9967). Numbers indicate character-states scored in the data matrix and the arrows indicate anterior direction. Scale bars equal 1 cm in (A, B), 5 mm in (C, D), and 1 mm in (E).

The ventral border of the orbit is gently concave in most diapsids, but in Garjainia prima (PIN 2394/5), Erythrosuchus africanus (BP/1/5207) and Youngosuchus sinensis (IVPP V3239) the dorsal margins of the anterior and ascending processes of the jugal form an acute angle between each other. As a result, the ventral border of the orbit is V-shaped in lateral view in these taxa.

95. Jugal, anterior extension of the anterior process: anterior to the level of mid-length of the orbit (0); up to or posterior to the level of mid-length of the orbit (1) (reworded from DeBraga & Rieppel, 1997; reworded from Müller, 2004: 128; modified and reworded from Ezcurra, Scheyer & Butler, 2014: 123).

The anterior process of the jugal forms most of the ventral border of the orbit in the vast majority of the diapsids sampled here. By contrast, the anterior process of the jugal is restricted to the posteroventral border of the orbit in early rhynchocephalians (e.g., Gephyrosaurus bridensis: Evans, 1980; Planocephalosaurus robinsonae: Fraser, 1982), Jesairosaurus lehmani (ZAR 08), and Trilophosaurus buettneri (Spielmann et al., 2008). In these taxa, the maxilla forms the anteroventral border of the orbit.

96. Jugal, participation of the anterior process in the border of the antorbital fenestra: present (0); absent, excluded by contact between the maxilla and lacrimal (1) (Clark, Sues & Berman, 2001; Ezcurra, Lecuona & Martinelli, 2010: 14; Nesbitt, 2011: 69). This character is not applicable in taxa that lack an antorbital fenestra or the anterior process of the jugal does not extend anteriorly to the level of mid-length of the orbit (Fig. 18).

See comments in Nesbitt (2011: character 69).

97. Jugal, longitudinal ridge or bump(s) on the lateral surface of the main body: absent (0); present (1) (Sereno & Novas, 1994; Nesbitt, 2011: 75; modified from Dilkes & Arcucci, 2012: 21) (Figs. 17–19).

See comments in Nesbitt (2011: character 75).

98. Jugal, multiple pits on the lateral surface of the main body: absent (0); present (1) (Dilkes, 1995; Butler et al., 2015: 17) (Fig. 17).

Although several archosauromorphs possess a flat lateral surface of the main body of the jugal, numerous species possess an ornamented jugal, with a tuberosity or tuber on its lateral surface (e.g., erythrosuchids: PIN 2394/5, Gower et al., 2014; Chanaresuchus bonapartei: PULR 07; Gracilisuchus stipanicicorum: MCZ 4117). However, among the sampled archosauromorphs, only in the early rhynchosaurs Mesosuchus browni (Dilkes, 1998), Howesia browni (Dilkes, 1995) and Eohyosaurus wolvaardti (Butler et al., 2015), Vancleavea campi (USNM 508519, cast of GR 138), and at least some specimens of Prolacerta broomi (BP/1/471, 4504a; SAM-PK-K10797), the lateral surface of the main body of the jugal possesses multiple circular pits.

99. Jugal, ascending process forming the entire anterior border of the infratemporal fenestra: absent (0); present, postorbital excluded from the anterior border of the infratemporal fenestra (1) (New character). This character is inapplicable in taxa in which the anterior process of the squamosal possesses an extensive contact with the postorbital and contacts the jugal, or lacks an infratemporal fenestra or an ascending process on the jugal (Fig. 17).

In most diapsids the ascending process of the jugal is restricted to the anteroventral border of the infratemporal fenestra. By contrast, the ascending process of the jugal possesses an extensive contact with the ventral process of the postorbital along all or most of its length and thus forms the entire anterior border of the infratemporal fenestra in the allokotosaurians Pamelaria dolichotrachela (ISI R316/1) and Azendohsaurus madagaskarensis (Flynn et al., 2010), the rhynchosaurs Eohyosaurus wolvaardti (SAM-PK-K10159), Rhynchosaurus articeps (NHMUK PV R1236, SHYMS 1), and Bentonyx sidensis (BRSUG 21200), Proterosuchus goweri (NMQR 880), “Chasmatosaurus” yuani (IVPP V4067), Tropidosuchus romeri (PVL 4606), and a referred specimen of Gualosuchus reigi (PVL 4576).

100. Jugal, length of the posterior process versus the height of its base: 0.49–1.27 (0); 1.59–3.77 (1); 4.07–5.37 (2) (modified from Parrish, 1992; Gower & Sennikov, 1997; Ezcurra, Lecuona & Martinelli, 2010: 98, cf. 150), ORDERED (Figs. 17 and 19).

This character describes the elongation of the posterior process of the jugal in relation to its height at the base.

101. Jugal posterior process with a distinct lateroventral orientation with respect to the sagittal axis of the snout: absent (0); present (1) (Butler et al., 2015: 19).

The main axis of the posterior process of the jugal is parallel to the longitudinal axis of the skull in most diapsids, but this process is distinctly lateroventrally oriented in rhynchosaurids (Butler et al., 2015) and Simoedosaurus lemoinei (Sigogneau-Russell & Russell, 1978). The lateral component of this process contributes to the transversely broad temporal region of rhynchosaurids.

102. Jugal, distal half of the posterior process: tapering (0); subrectangular (1) (New character). The character is inapplicable if the posterior process of the jugal is forked by the quadratojugal (Fig. 17).

The posterior process of the jugal tapers distally or bifurcates at its contact with the anterior process of the quadratojugal in most diapsids. However, the posterior process of the jugal possesses a squared distal tip, resulting in a subrectangular distal half of the process, in Proterosuchus fergusi (RC 846; SAM-PK-11208, K10603), Proterosuchus alexanderi (NMQR 1484), and “Chasmatosaurus” yuani (IVPP V4067).

103. Jugal, posterior process forms entirely or almost entirely the ventral border of the infratemporal fenestra (it also applies if the lower temporal bar is incomplete): absent (0); present (1) (New character). This character is inapplicable in taxa that lack an infratemporal fenestra (Fig. 19).

The posterior process of the jugal entirely or almost entirely forms the ventral border of the infratemporal fenestra (regardless of whether the lower temporal bar is complete or not) in Planocephalosaurus robinsonae (Fraser, 1982), proterosuchians (e.g., Proterosuchus fergusi: RC 846; SAM-PK-11208, K10603; Proterosuchus alexanderi: NMQR 1484; Proterosuchus goweri: NMQR 880; “Chasmatosaurus” yuani: IVPP V4067; Fugusuchus hejiapanensis: Cheng, 1980; Garjainia prima: PIN 2394/5; Erythrosuchus africanus: BP/1/5207; Shansisuchus shansisuchus: Young, 1964), Youngosuchus sinensis (IVPP V3239), Euparkeria capensis (SAM-PK-5867), and several archosaurs (e.g., Smilosuchus gregorii: UCMP 27200; Riojasuchus tenuisceps: PVL 3827; gracilisuchids: MCZ 4117, IVPP V3237; Prestosuchus chiniquensis: UFRGS-PV-0156-T; Heterodontosaurus tucki: Norman et al., 2011). By contrast, in most non-archosauriform archosauromorphs the jugal does not reach the posteroventral corner of the infratemporal fenestra because the process terminates anterior to this point or its posterior participation in the border of the infratemporal opening is excluded by the anterior process of the quadratojugal (e.g., Jesairosaurus lehmani: Jalil, 1997; Azendohsaurus madagaskarensis: Flynn et al., 2010; Mesosuchus browni: Dilkes, 1998).

104. Jugal, base of the posterior process with a semi-elliptical, ventral expansion in lateral view: absent (0); present (1) (reworded from Gower & Sennikov, 1997; Ezcurra, Lecuona & Martinelli, 2010: 97) (Fig. 24).

The putative proterosuchians Fugusuchus hejiapanensis (Cheng, 1980, unpublished photographs) and Sarmatosuchus otschevi (Gower & Sennikov, 1997) possess a ventrally expanded base of the posterior process of the jugal. A similar condition is also present in one referred specimen of Erythrosucus africanus (BP/1/5207) but not in other specimens (e.g., BP/1/3893). In the vast majority of diapsids, the base of the posterior process of the jugal is not semi-elliptical and it has a straight ventral margin.

105. Jugal, posterior process: lies dorsal to the anterior process of the quadratojugal (0); lies ventral to the anterior process of the quadratojugal (1); splits the anterior process of the quadratojugal (2); is splited by the anterior process of the quadratojugal (3) (Nesbitt, 2011: 71). This character is inapplicable to taxa that lack a quadratojugal or have an open lower temporal fenestra (Figs. 17, 19 and 20 of Nesbitt, 2011).

See comments in Nesbitt (2011: character 71).

106. Jugal, posterior termination of the posterior process: anterior to or at level with the posterior border of the infratemporal fenestra (0); posterior to the infratemporal fenestra (1) (Nesbitt, 2011: 72). This character is inapplicable in taxa that lack an infratemporal fenestra (Fig. 19).

See comments in Nesbitt (2011: character 72).

107. Prefrontal, contact its counterpart in the median line of the skull roof: absent (0); present (1) (Dilkes, 1998: 125).

The general condition in diapsids is the contact of the nasals and the frontals at approximately the level of the anterior border of the orbit on the skull roof. In choristoderans, the prefrontal contacts its counterpart on the median line of the skull roof, thus preventing the nasals from contact with the frontals (Sigogneau-Russell & Russell, 1978; Evans, 1990).

108. Prefrontal, suture with the nasal: parasagittal, at least in its posterior third, or anterolateral (0); anteromedial (1) (Laurin, 1991: E1; Reisz, Laurin & Marjanović, 2010: 33; modified from Ezcurra, Scheyer & Butler, 2014: 32). This character is inapplicable if the prefrontals meet each other in the median line.

109. Prefrontal, subtriangular medial process: absent, nasal-frontal suture transversely broad (0); present, nasal-frontal suture strongly transversely reduced (1) (New character) (Fig. 7).

The non-archosauriform archosauromorphs Tanystropheus longobardicus (Nosotti, 2007), Trilophosaurus buettneri (Spielmann et al., 2008), and Prolacertoides jimusarensis (IVPP V3233) possess a sub-triangular medial process on the prefrontal that strongly reduces the nasal-frontal suture on the skull roof. By contrast, the medial margin of the prefrontal is straight or slightly convex in dorsal view in other sampled diapsids.

110. Prefrontal, groove on the lateral surface of the main body opening into the orbital border: absent (0); present (1) (New character) (Fig. 17).

The erythrosuchids Garjainia prima (PIN 2394/5), Garjainia madiba (Gower et al., 2014), and Erythrosuchus africanus (Gower, 2003) differ from other archosauromorphs in the presence of a deep groove on the lateral surface of the main body of the prefrontal. This groove opens into the anterodorsal border of the orbit.

111. Prefrontal, lateral surface of the orbital margin: smooth or slight grooves present (0); rugose sculpturing present (1) (Nesbitt et al., 2015: 237) (Fig. 17).

See comments in Nesbitt et al. (2015: character 237).

112. Frontal, frontals fused to one another: absent (0); present (1) (Benton, 1985; Gauthier, 1986; Pritchard et al., 2015: 14) (Fig. 23).

The fusion between both frontals is a condition restricted to the rhynchocephalians Gephyrosaurus bridensis (Evans, 1980) and Planocephalosaurus robinsonae (Fraser, 1982) among the diapsids sampled here.

113. Frontal, suture with the nasal: transverse (0); oblique, forming an angle of at least 60°with long axis of the skull and frontals entering between both nasals (1); oblique and nasals entering considerably between frontals in a non-interdigitate suture (2) (DeBraga & Rieppel, 1997; Müller, 2004: 154; Nesbitt, 2011: 43, in part; modified from Ezcurra, Scheyer & Butler, 2014: 128). This character is inapplicable if the nasal is received by a slot in the frontal or the nasal does not contact the frontal (Fig. 23 and Fig. 18 of Nesbitt, 2011).

See comments in Nesbitt (2011: character 43).

114. Frontal, orbital border in mature individuals: absent or anteroposteriorly short (0); anteroposteriorly long and forms most of the dorsal edge of the orbit (1) (Reisz & Dilkes, 2003: 13; Reisz, Laurin & Marjanović, 2010: 26; Ezcurra, Scheyer & Butler, 2014: 25) (Fig. 23).

In some archosauromorphs the posterior process of the prefrontal and the anterior process of the postfrontal closely approach or even contact each other along the dorsal border of the orbit (e.g., Pamelaria dolichotrachela: ISI R316/1; Mesosuchus browni: SAM-PK-6536; Proterosuchus alexanderi: NMQR 1484; Erythrosuchus africanus: (Gower, 2003); Youngosuchus sinensis: IVPP V3239; Asperoris mnyama: NHMUK PV R36615; Batrachotomus kufferzellensis: Gower, 1999). This condition results in the restriction or exclusion of the frontal from the dorsal margin of the orbit.

115. Frontal, dorsal surface: flat or slightly depressed (0); with longitudinal ridge along midline (1) (Wu & Chatterjee, 1993; Nesbitt, 2011: 42) (Fig. 17 of Nesbitt, 2011).

See comments in Nesbitt (2011: character 42).

116. Frontal, suture with parietal: mostly transverse or parietals entering slightly between frontals on the median line, forming an obtuse-angled suture (0); parietals entering strongly between both frontals, forming an acute-angled suture (1); W-shaped suture (2) (Reisz & Dilkes, 2003: 4; Müller, 2004: 10; Reisz, Laurin & Marjanović, 2010: 27; modified from Ezcurra, Scheyer & Butler, 2014: 26; Pritchard et al., 2015: 16) (Figs. 8 and 23).

The sampled non-saurian diapsids (Petrolacosaurus kansensis: Reisz, 1981), Youngina capensis (BP/1/3859), Planocephalosaurus robinsonae (Fraser, 1982), Macrocnemus bassanii (PIMUZ T4822), Jesairosaurus lehmani (ZAR 07), Prolacerta broomi (BP/1/2675, UMCZ 2003.41R) and Kadimakara australiensis (QM F6710) possess a long posterolateral process of the frontals that results in an acute-angled, long suture with a strongly anteriorly tapering median projection formed by both parietals. In choristoderans (Cteniogenys sp.: (Evans, 1990); Simoedosaurus lemoinei: Sigogneau-Russell & Russell, 1978), Vancleavea campi (Nesbitt et al., 2009), Asperoris mnyama (Nesbitt, Butler & Gower, 2013), and Chanaresuchus bonapartei (PULR 07) the frontals form a W-shaped suture with the parietals as a result of a subdivided anterior end of the latter bones. In other saurians sampled here, the parietals extend slightly between the frontals along the midline, forming an obtuse-angled suture.

117. Frontal, participates on the anteromedial corner of the supratemporal fossa: absent (0); present (1) (Rieppel, Mazin & Tchernov, 1999; modified from (Müller, 2004): 178; modified from Ezcurra, Scheyer & Butler, 2014: 129). This character is inapplicable in taxa that lack a supratemporal fossa or fenestra.

Among the diapsids sampled here, the supratemporal fossa extends anteriorly onto the dorsal surface of the posterior end of the frontal only in the dinosaurs Heterodontosaurus tucki (Norman et al., 2011) and Herrerasaurus ischigualastensis (Sereno & Novas, 1994).

118. Frontal, dorsal surface adjacent to sutures with the postfrontal (if present) and parietal: flat to slightly concave (0); possesses a longitudinal and deep depression (1) (modified from Dilkes, 1998: 20; Ezcurra, Scheyer & Butler, 2014: 130; Pritchard et al., 2015: 17, in part) (Fig. 16).

Dilkes (1998) proposed that the presence of a longitudinal and deep depression on the dorsal surface of the posterior end of the frontal was a synapomorphy of Rhynchosauria. In addition, this condition is also present in the allokotosaurian Azendohsaurus madagaskarensis (UA-7-20-99-653). In other diapsids, the dorsal surface of the posterior end of the frontal is flat or slightly concave.

119. Frontal, longitudinal groove: longitudinally extended along most of the surface of the frontal (0); anterolaterally-to-posteromedially extended along the posterior half of the frontal (1) (Butler et al., 2015: 24). This character is inapplicable in taxa that lack a longitudinal depression with deep pits on the frontal (Fig. 16).

Among the sampled species that possess a longitudinal and deep depression on the dorsal surface of the posterior end of the frontal, only in Azendohsaurus madagaskarensis (UA-7-20-99-653), Rhynchosaurus articeps (NHMUK PV R1236, R1237, SHYMS 1), and Bentonyx sidensis (BRSUG 21200) this groove is anterolaterally-to-posteromedially extended along the posterior half of the dorsal surface of the frontal.

120. Frontal, ventral surface: hourglass-shaped median longitudinal canal for the passage of the olfactory tract and olfactory bulb moulds on the anterior end of the bone (0); median longitudinal canal for the passage of the olfactory tract only slightly constricted, no olfactory bulb moulds and distinct semilunate posteromedially-to-anterolaterally oriented ridge on the skull roof, extending onto the prefrontal (1) (New character) (Fig. 23).

The ventral surface of the anterior end of the frontal possesses a suboval, moderately well-defined concavity that houses the olfactory bulbs of the anterior brain in most diapsids. However, in the erythrosuchids Garjainia prima (Huene, 1960), Erythrosuchus africanus (NHMUK PV R3592, NMQR 1473) and Shansisuchus shansisuchus (Young, 1964) there is a median longitudinal canal for the passage of the olfactory tract, but there are no olfactory bulb impressions and the distinct semilunate posteromedially-to-anterolaterally oriented ridges that define these impressions on the skull roof.

121. Frontal, olfactory tract on the ventral surface of the frontal: maximum transverse constriction point well posterior to the moulds of the olfactory bulbs and posterolateral margin of the bulbs delimited by a low ridge (0); maximum transverse constriction of the olfactory tract immediately posterior to the moulds of the olfactory bulbs and posterolateral margin of the bulbs well delimited by a thick, tall ridge (1) (New character). This character is inapplicable in taxa that lack olfactory bulb moulds and constriction of the olfactory tract canal (Fig. 23).

The tanystropheids Macrocnemus bassanii (PIMUZ T2472) and Tanystropheus longobardicus (Nosotti, 2007: Figs. 42 and 44) share the presence of a maximum constriction point of the olfactory tract placed immediately posterior of the moulds of the olfactory bulbs and a posterolateral margin of the bulbs well delimited by a thick ridge. The condition in Amotosaurus rotfeldensis could not be determined in the currently available and prepared specimens. By contrast, in other diapsids sampled here the maximum transverse constriction point of the olfactory tract is well posterior to the impressions of the olfactory bulbs and the posterolateral margin of the bulbs is delimited by a faint change in slope (e.g., Planocephalosaurus robinsonae: NHMUK PV R9975; Tasmaniosaurus triassicus: UTGD 54655; Sarmatosuchus otschevi: PIN 2865/68).

122. Postfrontal: equivalent in size to postorbital (0); reduced to approximately less than half the size of the postorbital (1); absent (2) (Benton, 1985; Nesbitt, 2011: 44, in part; Ezcurra, Lecuona & Martinelli, 2010: 10; Dilkes & Arcucci, 2012: 16, in part; Pritchard et al., 2015: cf. 15, in part), ORDERED (Fig. 17).

See comments in Nesbitt (2011: character 44).

123. Postfrontal, participation in the border of the supratemporal fenestra: absent (0); present (1) (modified from Laurin, 1991: A1, B3; Dilkes, 1998: 22; Müller, 2004: 89, 90; Reisz, Laurin & Marjanović, 2010: 39; Ezcurra, Scheyer & Butler, 2014: 38, in part; Pritchard et al., 2015: 27). Scored as inapplicable in taxa that lack a postfrontal bone (Fig. 16).

The postfrontal (if present) is excluded from the anterior border of the supratemporal fenestra by the contact between the ascending process of the postorbital and the parietal in most archosauromorphs. However, the postfrontal participates in the border of the supratemporal opening and prevents a postorbital-parietal contact in Youngina capensis (Gow, 1975), Gephyrosaurus bridensis (Evans, 1980), Planocephalosaurus robinsonae (Fraser, 1982), Tanystropheus longobardicus (Nosotti, 2007), and Jesairosaurus lehmani (ZAR 07).

124. Postfrontal, shape of dorsal surface: flat or slightly concave towards raised orbital rim (0); depression with deep pits (1) (Dilkes, 1998: 21; Ezcurra, Scheyer & Butler, 2014: 131; Pritchard et al., 2015: 17, in part). Scoring inapplicable in taxa that lack a postfrontal (Fig. 16).

The dorsal surface of the postfrontal is flat or slightly concave in most diapsids, but in the rhynchosaurs Mesosuchus browni (Dilkes, 1998), Howesia browni (Dilkes, 1995), and Bentonyx sidensis (BRSUG 21200) and in the phytosaur Parasuchus hislopi (ISI R42) the dorsal surface of the bone is invaded by a depression that lodges deep pits within it.

125. Postorbital-jugal, postorbital bar: composed by both jugal and postorbital in nearly equal proportion (0); composed mostly by the postorbital (1) (Nesbitt, 2011: 67).

See comments in Nesbitt (2011: character 67).

126. Postorbital-squamosal, upper temporal bar: located approximately at level of mid-height of the orbit (0); located approximately aligned to the dorsal border of the orbit (1) (New character) (Figs. 17 and 19).

In early diapsids, including most basal archosauromorphs, the upper temporal bar is located approximately at level of the mid-height of the orbit and, as a result, the supratemporal fossa is broadly visible in lateral view. By contrast, in Youngina capensis (BP/1/375, TM 3603), rhynchosaurs (e.g., Mesosuchus browni: SAM-PK-6536; Rhynchosaurus articeps: NHMUK PV R1236, SHYMS 1) and archosauriforms (e.g., Proterosucus fergusi: SAM-PK-11208, K10603, RC 846; Erythrosuchus africanus: BP/1/5207; Euparkeria capensis: SAM-PK-5867; Tropidosuchus romeri: PVL 4606; Parasuchus hislopi: ISI R42; Gracilisuchus stipanicicorum: MCZ 4117; Herrerasaurus ischigualastensis: PVSJ 407) the upper temporal bar is placed at level with the dorsal border of the orbit and the supratemporal fenestra faces dorsally and is not visible in lateral view.

127. Postorbital-squamosal, contact: restricted to the dorsal margin of the elements (0); continues ventrally for much or most of the ventral length of the squamosal, but squamosal does not contact jugal (1); continues ventrally for much or most of the ventral length of the squamosal and squamosal contacts jugal (2) (modified from Nesbitt, 2011: 66), ORDERED (Figs. 17 and 19 of Nesbitt, 2011).

See comments in Nesbitt (2011: character 66).

128. Postorbital, lateral boss adjacent to orbital margin: absent (0); present (1) (Reisz & Dilkes, 2003: 37; Reisz, Laurin & Marjanović, 2010: 44; Ezcurra, Scheyer & Butler, 2014: 43) (Fig. 19).

The lateral surface of the anterior process of the postorbital possesses a lateral, rounded or gently pointing prominence adjacent to the orbital border of the bone in some archosauriforms, including erythrosuchids (e.g., Garjainia madiba: Gower et al., 2014; Erythrosuchus africanus: BP/1/5207), Youngosuchus sinensis (IVPP V3239), Ornithosuchus longidens (Walker, 1964), gracilisuchids (e.g., Gracilisuchus stipanicicorum: MCZ 4177), and loricatans (e.g., Prestosuchus chiniquensis: UFRGS-PV-0156-T).

129. Postorbital, supratemporal fossa extending onto the ascending process: absent (0); present (1) (Gauthier, 1986; reworded from Nesbitt, 2011: 144). This character is not applicable in taxa that lack a supratemporal fossa medially to the supratemporal fenestra or the supratemporal fenestra (Figs. 18 and 19 of Nesbitt, 2011).

See comments in Nesbitt (2011: character 144).

130. Postorbital, posterior process extends close to or beyond the level of the posterior margin of the supratemporal fenestrae: absent (0); present (1) (Laurin, 1991: I1; Reisz & Dilkes, 2003: 23; Müller, 2004: 131; reworded from Reisz, Laurin & Marjanović, 2010: 45; modified from Ezcurra, Scheyer & Butler, 2014: 44; Pritchard et al., 2015: cf. 28) (Fig. 17).

This character describes the degree of posterior development of the posterior process of the postorbital. The presence of an elongated posterior process of the postorbital that extends close to or beyond the level of the posterior border of the supratemporal fenestra is present in Youngina capensis (BP/1/3859) and multiple, disparate archosauromorph species. This condition does not seem to have a priori a clear phylogenetic signal, but it may be informative at low taxonomic levels.

131. Postorbital, extension of the ventral process: ends much higher than the ventral border of the orbit (0); ends close to or at the ventral border of the orbit (1) (Desojo, Ezcurra & Schultz, 2011: 112) (Fig. 17).

In the vast majority of the diapsids sampled here the ventral process of the postorbital finishes well dorsal to the ventral border of the orbit. However, in some tanystropheids (e.g., Macrocnemus bassanii: PIMUZ T4822), Pamelaria dolichotrachela (ISI R316/1), Azendohsaurus madagaskarensis (UA-7-20-99-653), some proterochampsids (e.g., Tropidosuchus romeri: PVL 4606; Cerritosaurus binsfeldi: CA s/n), and phytosaurs (e.g., Parasuchus hislopi: ISI R42) the ventral process of the postorbital finishes close to or reaches the base of the ascending process of the jugal, thus excluding the latter bone entirely or almost entirely from participation in the posterior margin of the orbit.

132. Postorbital, ventral process in lateral view: continuously anteriorly curved or straight (0); distinctly anteriorly flexed (1) (New character).

In some archosaurs included in the present analysis (i.e., Turfanosuchus dabanensis: IVPP V3237; Batrachotomus kupferzellensis: SMNS 80260; Silesaurus opolensis: ZPAL AbIII/1930/92/25) the ventral process of the postorbital possesses a distinct anterior flexure between two regions that have straights posterior margins. By contrast, the ventral process of the postorbital curves continuously anteriorly or is straight in lateral view in the vast majority of diapsids.

133. Postorbital, depression on the lateral surface of the ventral process: absent (0); present (1) (Wu, Liu & Li, 2001; Ezcurra, Lecuona & Martinelli, 2010: 17; Dilkes & Arcucci, 2012: 24) (Figs. 16 and 18).

See comments in Dilkes & Arcucci (2012: character 24).

134. Postorbital, anteriorly projecting, rounded spur on the anterior edge of the ventral process indicating the lower delimitation of the eyeball: absent (0); present (1) (Benton & Clark, 1988; Rauhut, 2003) (Fig. 17).

The anterior margin of the ventral process of the postorbital possesses a low, rounded prominence that enters into the orbital cavity in some erythrosuchids (e.g., Garjainia prima: PIN 2394/5; Shansisuchus shansisuchus: Wang et al., 2013), some proterochampsids (e.g., Gualosuchus reigi: PVL 4576; Pseudochampsa ischigualastensis: Trotteyn & Ezcurra, 2014), Prestosuchus chiniquensis (UFRGS-PV-0156-T), and Lewisuchus admixtus (PULR 01). A similar, but generally better-developed condition is usually present in some large-bodied theropod clades, such as abelisaurids and carcharodontosaurids (Rauhut, 2003; Carrano, Sampson & Forster, 2002).

135. Squamosal, completely covering the quadrate in lateral view: present (0); absent (1) (Gauthier, 1984; Benton, 1985; Pritchard et al., 2015: 34).

The squamosal has a plate-like, anteroposteriorly broad ventral flange that completely covers the quadrate in lateral view in the early diapsid Petrolacosaurus kansensis (Reisz, 1981). By contrast, in other diapsids sampled here, the squamosal possesses a considerably anteroposteriorly narrower ventral process that allows the exposition of the proximal half of the quadrate in lateral view.

136. Squamosal, overhanging quadrate laterally: absent (0); present (1) (Benton, 2004; Ezcurra, Lecuona & Martinelli, 2010: 15). This character is inapplicable in taxa that the quadrate is completely covered by the squamosal in lateral view.

The anterior process and main body of the squamosal possess a strong lateral development that overhangs laterally the quadrate in the choristoderan Simoedosaurus lemoinei (Sigogneau-Russell & Russell, 1978) and the suchian Gracilisuchus stipanicicorum (MCZ 4117, PULR 08). In other diapsids sampled here, the lateral surface of the squamosal is placed at approximately the same level as the lateral surface of the quadrate and quadratojugal.

137. Squamosal, anterior process forms more than half of the lateral border of the supratemporal fenestra: absent (0); present (1) (New character). This character is inapplicable in taxa lacking a supratemporal fenestra (Fig. 16).

The anterior process of the squamosal has an extensive longitudinal suture with the postorbital and reaches the base of the ascending process of the latter bone in several diapsids (e.g., Youngina capensis: BP/1/3859; Trilophosaurus buettneri: Spielmann et al., 2008; Prolacerta broomi: UMZC 2003.41R; Proterosuchus alexanderi: NMQR 1484; Chanaresuchus bonapartei: PULR 07; Parasuchus angustifrons: BSPG 1931 X 502; Prestosuchus chiniquensis: UFRGS-PV-0156-T). This condition excludes the postorbital from the posterior half of the lateral border of the supratemporal fenestra.

138. Squamosal, anteroventral process: absent (0); present (1) (Reisz & Dilkes, 2003: 8; Reisz, Laurin & Marjanović, 2010: 35; Nesbitt, 2011: 52; Ezcurra, Scheyer & Butler, 2014: 34, in part) (Figs. 18 and 19 of Nesbitt, 2011).

See comments in Nesbitt (2011: character 52).

139. Squamosal, transition between the anterior and ventral processes: sharp, posterodorsal border of the infratemporal fenestra with square outline (0); gentle, widely rounded posterodorsal border of the infratemporal fenestra (1) (New character). This character is inapplicable in taxa that lack a ventral process of the squamosal (Figs. 8, 17, 18 and 24).

In several putative proterosuchians, the anterior and ventral processes of the squamosal form a wide angle between each other in lateral view (e.g., Proterosuchus fergusi: RC 846, SAM-PK-11208, K10603; Proterosuchus goweri: NMQR 880; Proterosuchus alexanderi: NMQR 1484; “Chasmatosaurus” yuani: IVPP V4067; Fugusuchus hejiapanensis: (Cheng, 1980); Sarmatosuchus otschevi: PIN 2865/68; Garjainia prima: PIN 2394/5). This condition results in a broadly concave posterodorsal corner of the infratemporal fenestra and is also present in a variety of diapsids. By contrast, the anterior and ventral processes of the squamosal meet at a right angle and form a squared posterodorsal corner of the infratemporal fenestra in other diapsids.

140. Squamosal medial process: short, forming approximately half or less of the posterior border of the supratemporal fenestra (0); long, forming entirely or almost entirely the posterior border of the supratemporal fenestra (1) (Butler et al., 2015: 40) (Fig. 16).

This character describes the degree of extension of the medial process of the squamosal and its contribution to the posterior border of the supratemporal fenestra. The medial process of the squamosal forms most of the posterior border of the supratemporal opening in some saurians sampled here (e.g., Rhynchosaurus articeps: SHYMS 1, NHMUK PV R1237; Bentonyx sidensis: BRSUG 21200; Vancleavea campi: Nesbitt et al., 2009; Doswellia kaltenbachi: USNM 214823).

141. Squamosal, posterior process length: does not extend posterior to the head of the quadrate (0); extends posterior to the head of the quadrate (1) (Nesbitt et al., 2009; Ezcurra, Lecuona & Martinelli, 2010: 157; Nesbitt, 2011: 48; Dilkes & Arcucci, 2012: 19; Pritchard et al., 2015: cf. 35). This character is inapplicable in taxa where the quadrate is completely covered by the squamosal in lateral view (Figs. 18, 19 and 24).

See comments in Nesbitt (2011: character 48) and Dilkes & Arcucci (2012: character 19).

142. Squamosal, posterior process shape: straight (0); ventrally curved (1) (Ezcurra, Lecuona & Martinelli, 2010: 165). This character is inapplicable in taxa where the quadrate is completely covered by the squamosal in lateral view (Figs. 18 and 24).

The posterior process of the squamosal curves ventrally in lateral view in some proterochampsids (Cerritosaurus binsfeldensis: CA s/n; Gualosuchus reigi: PVL 4576; Chanaresuchus bonapartei: MCZ 4037, 4039, PVL 4586), phytosaurs (e.g., Smilosuchus gregorii: UCMP 27200), pseudosuchians (e.g., Turfanosuchus dabanensis: IVPP V3237; Batrachotomus kupferzellensis: SMNS 52970), and the ornithischian Heterodontosaurus tucki (SAM-PK-K337, K1332).

143. Squamosal, ventral process: present (0); absent (1) (Gauthier, 1984; Benton, 1985; Pritchard et al., 2015: 33).

The squamosal lacks a ventral process and, as a result, lacks the typical triradiate morphology present in the non-archosauriform archosauromorphs Tanystropheus longobardicus (Wild, 1973; Nosotti, 2007) and Trilophosaurus buettneri (Spielmann et al., 2008). In other diapsids sampled here, the ventral process of the squamosal is present and extends ventrally or anteroventrally, forming the dorsal portion of the posterior border of the infratemporal fenestra.

144. Squamosal, ventral process shape: anteroposteriorly broad and plate-like (0); anteroposteriorly narrow and strap-like (1) (Gauthier, 1986; Benton & Clark, 1988; Laurin, 1991: D2, E4; Reisz & Dilkes, 2003: 24; modified from Dilkes, 1998: 34 and Reisz, Laurin & Marjanović, 2010: 37; Ezcurra, Lecuona & Martinelli, 2010: 151, in part; Nesbitt, 2011: 56, in part; modified from Ezcurra, Scheyer & Butler, 2014: 36, in part). This character is inapplicable in taxa that lack a ventral process in the squamosal (Figs. 17, 18 and 24).

See comments in Nesbitt (2011: character 56).

145. Squamosal, ventral process orientation: posteroventrally directed, vertical, or more than 45°from the vertical (0); anteroventrally directed at 45°or less (1) (modified from Ezcurra, Lecuona & Martinelli, 2010: 167). This character is inapplicable in taxa that lack a ventral process in the squamosal (Fig. 18).

The ventral process of the squamosal is posteroventrally or ventrally oriented in lateral view in most diapsids. By contrast, the ventral process of the squamosal is anteroventrally oriented in lateral view in Paliguana whitei (AM 3585), Nicrosaurus kapffi (NHMUK PV R42743), Smilosuchus gregorii (UCMP 27200), the gracilisuchids Turfanosuchus dabanensis (IVPP V3237) and Gracilisuchus stipanicicorum (MCZ 4117, PULR 08), and Heterodontosaurus tucki (SAM-PK-K337, K1332).

146. Squamosal, contribution of the ventral process to the posterior border of the infratemporal fenestra: forms less than half of the border of the fenestra (0); forms more than half of the border, but quadratojugal or quadrate broadly participates in the border of the fenestra (1); forms almost completely the border of the fenestra (2) (New character), ORDERED. This character is inapplicable in taxa that lack a ventral process on the squamosal (Fig. 19).

This character describes the degree of ventral development of the ventral process of the squamosal and its contribution to the posterior border of the infratemporal fenestra. The states of this character are rather variable among the diapsids sampled here, but only in the early neodiapsids Acerosodontosaurus piveteaui (MNHN 1908-32-57) and Youngina capensis (TM 3603) and the proterosuchians “Chasmatosaurus” yuani (IVPP V4067), Erytrhosuchus africanus (BP/1/5207), and Shansisuchus shansisuchus (Young, 1964) the ventral process of the squamosal forms almost completely the posterior border of the infratemporal opening.

147. Squamosal, posterodorsally-to-anteroventrally oriented tuck on the lateral surface of the ventral process: absent (0); present (1) (New character). This character is inapplicable in taxa that lack a ventral process in the squamosal (Figs. 19 and 24).

The ventral process of the squamosal possesses a distinct tuck that results in a posterodorsally-to-anteroventrally oriented shelf on the lateral surface of the bone in the rhynchosaur Eohyosaurus wolvaardti (SAM-PK-K10159) and the erythrosuchids Erythrosuchus africanus (BP/1/5207; Gower, 2003) and Shansisuchus shansisuchus (Young, 1964). By contrast, the lateral surface of the ventral process of the squamosal is mostly flat in other diapsids sampled here.

148. Squamosal, longitudinal ridge on the lateral surface of the ventral process: absent (0); present (1) (Nesbitt, 2011: 51). This character is inapplicable in taxa that lack a ventral process in the squamosal (Fig. 18 of Nesbitt, 2011).

See comments in Nesbitt (2011: character 51).

149. Squamosal, posterodorsal portion with a supratemporal fossa: absent (0); present (1) (Nesbitt, 2011: 55). This character is inapplicable in taxa lacking a supratemporal fenestra (Fig. 16 of Nesbitt, 2011).

See comments in Nesbitt (2011: character 55).

150. Quadratojugal: absent or fused to the quadrate (0); present (1) (modified from Reisz & Dilkes, 2003: 36; Reisz, Laurin & Marjanović, 2010: 46; Ezcurra, Scheyer & Butler, 2014: 45; Pritchard et al., 2015: 38) (Fig. 24).

The quadratojugal is developed as a rather small, splint-like bone in several non-archosauriform saurians (e.g., Azendohsaurus madagaskarensis: Flynn et al., 2010; Mesosuchus browni: SAM-PK-6536; Prolacerta broomi: BP/1/471, SAM-PK-K10797) and species of Proterosuchus (Ezcurra & Butler, 2015a) and “Chasmatosaurus” yuani (IVPP V4067). However, the quadratojugal is absent or fused to the quadrate only in Planocephalosaurus robinsonae (Fraser, 1982), some specimens of Gephyrosaurus bridensis (Evans, 1980), and Tanystropheus longobardicus (Wild, 1973; Nosotti, 2007) among the diapsids sampled here.

151. Quadratojugal, shape: L-shaped or strip-like bone (0); subtriangular (1) (Sereno, 1991; Nesbitt, 2011: 46). This character is inapplicable in taxa that lack a quadratojugal or infratemporal fenestra (Fig. 8, Fig. 16 of Nesbitt, 2011).

See comments in Nesbitt (2011: character 46).

152. Quadratojugal-jugal, infratemporal fossa marked by a sharp edge: absent (0); present (1) (Nesbitt et al., 2009; Nesbitt, 2011: 47, in part; Dilkes & Arcucci, 2012: 23 and 25). This character is inapplicable in taxa that lack a quadratojugal or an infratemporal fenestra (Fig. 16 of Nesbitt, 2011).

See comments in Nesbitt (2011: character 47) and Dilkes & Arcucci (2012: character 23).

153. Quadratojugal, anterior process: absent, anteroventral margin of the bone rounded (0); incipient, short anterior prong on the anteroventral margin of the bone (1); distinctly present, in which the lower temporal bar is complete, but process terminates well posterior to the base of the posterior process of the jugal (2); distinctly present, in which the lower temporal bar is complete and participates in the posteroventral border of the infratemporal fenestra, and process terminates close to the base of the posterior process of the jugal (3) (Laurin, 1991: A2; Reisz & Dilkes, 2003: 3 and 9 and 11, in part; Reisz, Laurin & Marjanović, 2010: 40; Ezcurra, Lecuona & Martinelli, 2010: 1,18; Nesbitt, 2011: 70; Dilkes & Arcucci, 2012: 22, in part; modified from Ezcurra, Scheyer & Butler, 2014: 39, and Pritchard et al., 2015: cf. 32, 39), ORDERED. This character is inapplicable in taxa that lack an infratemporal fenestra or quadratojugal (Figs. 17 and 19).

See comments in Nesbitt (2011: character 70).

154. Quadratojugal, widely concave notch on the anterior margin of the ascending process: absent (0); present (1) (reworded from Dilkes & Arcucci, 2012: 26). This character is inapplicable in taxa that lack a quadratojugal, an anterior process of the quadratojugal or the infratemporal fenestra (Fig. 18).

See comments in Dilkes & Arcucci (2012: character 26).

155. Quadratojugal, depression along the posterior half of the ascending process up to the exposed lateral surface of its distal tip: absent (0); present (1) (New character) (Fig. 19).

The lateral surface of the ascending process of the quadratojugal is generally anteroposteriorly convex or flat in the vast majority of diapsids. However, in the erythrosuchids Erythrosuchus africanus (BP/1/5207) and Shansisuchus shansisuchus (Young, 1964) the posterior half of this process is extensively invaded by a depression with a gently anteroposteriorly concave surface.

156. Quadratojugal, posterior extension of the ventral end: absent, without a posteriorly arched quadratojugal (0); limited, ventral condyles of the quadrate broadly visible in lateral view (1); strongly developed, overlapping completely or almost completely the ventral condyles of the quadrate in lateral view (2) (New character), ORDERED. This character is inapplicable in taxa that lack a quadratojugal (Fig. 18).

The quadratojugal is not posteriorly arched in Youngina capensis (SAM-PK-K7578) and Gephyrosaurus bridensis (Evans, 1980), whereas this bone is bowed, with a distinct posterior extension of its ventral end that overlaps at least partially the distal condyles of the quadrate in most other diapsids sampled here. The second and third character-states describe the degree of posterior development of the ventral end of the quadratojugal and its overlap of the ventral condyles of the quadrate in lateral view. In the vast majority of rhynchosaurs and archosauriforms, the ventral condyles of the quadrate are completely or almost completely overlapped by the quadratojugal in lateral view.

157. Supratemporal: broad element (0); slender, in parietal and squamosal trough (1); absent (2) (Dilkes, 1998; Reisz & Dilkes, 2003: 22; Müller, 2004: 21; Reisz, Laurin & Marjanović, 2010: 52; Ezcurra, Lecuona & Martinelli, 2010: 13, in part; Nesbitt, 2011: 145, in part; Dilkes & Arcucci, 2012: 18, in part; Ezcurra, Scheyer & Butler, 2014: 51; Pritchard et al., 2015: 36), ORDERED (Fig. 17).

See comments in Nesbitt (2011: character 145).

158. Supratemporal, bifurcated medial border, in which a ventromedial process extends underneath the posterolateral process of the parietal: present (0); absent (1) (Butler et al., 2015: 44). This character is inapplicable in taxa lacking a supratemporal.

The supratemporal tapers medially in the vast majority of diapsids (if the bone is present), but in the early rhynchosaurs Mesosuchus browni (SAM-PK-6536) and Eohyosaurus wolvaardti (SAM-PK-K10159) the bone bifurcates medially. The ventral bifurcation of the supratemporal extends ventrally to the posterolateral process of the parietal in the latter two species.

159. Parietal, median contact between both parietals: suture present (0); fused with loss of suture (1) (Dilkes, 1998: 25; Nesbitt, 2011: 58; Ezcurra, Scheyer & Butler, 2014: 133; Pritchard et al., 2015: 19) (Fig. 16).

See comments in Nesbitt (2011: character 58).

160. Parietal, extension over interorbital region: absent or slight (0); present (1) (Reisz & Dilkes, 2003: 16; Reisz, Laurin & Marjanović, 2010: 28; Ezcurra, Scheyer & Butler, 2014: 27) (Figs. 6 and 23).

This character describes the anterior extension of the parietals in relation to the level of the posterior border of the orbit. The anterior margin of the parietals extends anteriorly over the interorbital region in Petrolacosaurus kansensis (Reisz, 1981), Acerosodontosaurus piveteaui (Currie, 1980), Youngina capensis (BP/1/3859), Macrocnemus bassanii (PIMUZ T4822), Tanystropheus longobardicus (Wild, 1973), Jesairosaurus lehmani (ZAR 07), Prolacerta broomi (BP/1/2675), Kadimakara australiensis (QM F6710), some specimens of Proterosuchus fergusi (RC 59), Euparkeria capensis (SAM-PK-5867), ornithosuchids (e.g., Ornithosuchus longidens: Walker, 1964; Riojasuchus tenuisceps: Bonaparte, 1971), and Gracilisuchus stipanicicorum (MCZ 4117).

161. Parietal, supratemporal fossa medial to the supratemporal fenestra: well exposed in dorsal view and mainly dorsally or dorsolaterally facing (0); poorly exposed in dorsal view and mainly laterally facing (1) (New character). This character is inapplicable in taxa that lack a supratemporal fossa (Fig. 16).

This character distinguishes between a mainly dorsally or dorsolaterally facing supratemporal fossa immediately lateral to the supratemporal fenestra and a dorsally poorly exposed and mainly laterally facing fossa. Ezcurra & Butler (2015a) discussed the presence of a polymorphic condition for this character in the available samples of Prolacerta broomi and Proterosuchus fergusi. Both character-states are present in disparate groups of diapsids and are probably highly homoplastic, but they may be phylogenetically informative at low taxonomic levels.

162. Parietal, pineal fossa on the median line of the dorsal surface: absent (0); present (1) (Parrish, 1992; Gower & Sennikov, 1997; Ezcurra, Lecuona & Martinelli, 2010: 100). The character should not be scored for early juveniles (Fig. 8).

The pineal fossa is a sub-circular, oval, or sub-rectangular depression, usually with a longitudinal main axis, that invades the dorsal surface of the parietals and sometimes the posterior end of the frontals on the median line of the skull roof. This fossa is present in Kadimakara australiensis (QM F6710), Proterosuchus fergusi (TM 201, SAM-PK-K1603), Proterosuchus goweri (NMQR 880), Proterosuchus alexanderi (NMQR 1484) and some erythrosuchids (e.g., Garjainia prima: PIN 2394/5; Erythrosuchus africanus: BP/1/5207, NHMUK PV R3592, NMQR 1473; Shansisuchus shansisuchus: IVPP V2501, V2504, V2505). Ezcurra & Butler (2015b) recently discussed the ontogenetic variation in the presence of a pineal fossa in the South African proterosuchid Proterosuchus fergusi, which is absent in juvenile individuals.

163. Parietal, position of the pineal fossa: restricted to the parietal (0); extended along frontal and parietal (1) (New character). This character is inapplicable in taxa lacking a pineal fossa (Figs. 8 and 23).

The pineal fossa extends along the frontals and parietals in Proterosuchus fergusi (TM 201, SAM-PK-K1603), Proterosuchus goweri (NMQR 880), Proterosuchus alexanderi (NMQR 1484) and the erythrosuchids Garjainia prima (PIN 2394/5) and Erythrosuchus africanus (BP/1/5207, NHMUK PV R3592, NMQR 1473).

164. Parietal, pineal foramen in dorsal view: large (0); reduced to a small, circular pit (1); absent (2) (modified from Laurin, 1991: G3; Müller, 2004: 12; Reisz, Laurin & Marjanović, 2010: 31; Ezcurra, Lecuona & Martinelli, 2010: 11, in part; Nesbitt, 2011: 63, in part; Ezcurra, Scheyer & Butler, 2014: 30; Pritchard et al., 2015: 22) (Figs. 6 and 8).

See comments in Nesbitt (2011: character 63).

165. Parietal, position of the pineal foramen in dorsal view: completely enclosed by parietals in the anterior half of the bone (excluding posterolateral processes and anterior projections of the parietals if present) (0); completely enclosed by parietals close to mid-length or in the posterior half of the bone (excluding posterolateral processes and anterior projections of the parietals if present) (1); enclosed by both frontals and parietals (2) (Reisz & Dilkes, 2003: 17; Müller, 2004: 12; Reisz, Laurin & Marjanović, 2010: 32; Ezcurra, Lecuona & Martinelli, 2010: 11, in part; modified from from Ezcurra, Scheyer & Butler, 2014: 31, and Pritchard et al., 2015: 23). Scored as inapplicable in taxa that lack a pineal foramen (Figs. 6 and 8).

The pineal foramen is a sub-circular opening that pierces the skull roof in the temporal region. This foramen is completely enclosed by the parietals and placed close to the mid-length or in the posterior half of these bones in the non-archosauromorph diapsids scored here (e.g., Petrolacosaurus kansensis: Reisz, 1981; Planocephalosaurus robinsonae: NHMUK PV R9963). By contrast, the pineal foramen is placed on the anterior half of the parietals in most archosauromorphs (e.g., Mesosuchus browni: SAM-PK-6536; Kadimakara australiensis: QM F6710; Proterosuchus fergusi: BP/1/3993; Proterosuchus goweri: NMQR 880), but it pierces the skull roof between both frontals and parietals in Azendohsaurus madagaskarensis (Flynn et al., 2010) and some specimens of Tanystropheus longobardicus (Nosotti, 2007).

166. Parietal, distinct transverse emargination adjacent to the posterior margin of the bone in late ontogeny: absent (0); present (1) (Müller, 2004: 177; Ezcurra, Scheyer & Butler, 2014: 134) (Fig. 8).

In the choristoderan Simoedosaurus lemoinei (Sigogneau-Russell & Russell, 1978) and archosauromorphs (Müller, 2004) the posterolateral process of the parietal possesses a sharp dorsal flange that defines the posterior border of the supratemporal fenestra. The posterior emargination of the parietals is absent in disparate archosauromorph species, namely Jesairosaurus lehmani (ZAR 07), Vancleavea campi (USNM 508519, cast of GR 138), Doswellia kaltenbachi (USNM 214823) and Riojasuchus tenuisceps (PVL 3827).

167. Parietal, posterolateral process: nearly vertical (0); ventrally inclined greater than 45°(1) (Heckert & Lucas, 1999; Nesbitt, 2011: 62) (Fig. 17 of Nesbitt, 2011).

See comments in Nesbitt (2011: character 62).

168. Parietal, posterolateral process height: dorsoventrally low, usually considerably lower than the supraoccipital (0); dorsoventrally deep, being plate-like in occipital view and subequal to the height of the supraoccipital (1) (New character) (Fig. 27).

The posterolateral process of the parietal is dorsoventrally deep, being subequal in height to the supraoccipital in the vast majority of archosauromorphs scored here to the exclusion of Protorosaurus speneri (BSPG 1995 I 5, cast of WMsN P47361), Tanystropheus longobardicus (Wild, 1973), Jesairosaurus lehmani (ZAR 08) and rhynchosaurids (e.g., Rhynchosaurus articeps: NHMUK PV R1236, SHYMS 1; Bentonyx sidensis: BRSUG 21200).

169. Parietal, posterolateral process with a strongly transversely convex dorsal margin elevated from the median line of the posterior margin of the skull roof: absent (0); present (1) (New character) (Fig. 23).

The dorsal margin of the posterolateral process of the parietal is straight or gently transversely convex in occipital view in the vast majority of diapsids scored here. By contrast, in the choristoderan Simoedosaurus lemoinei (Sigogneau-Russell & Russell, 1978), erythrosuchids except Shansisuchus shansisuchus (Young, 1964), and Youngosuchus sinensis (IVPP V3239) the posterolateral process of the parietal is wing-like in occipital view, with a strongly convex and elevated dorsal margin.

170. Parietal, tuberosity on the posterior surface of the base of the posterolateral process: absent (0); present (1) (New character) (Fig. 23).

Gower (2003) described the presence of a pair of tuberosities placed on the posterior surface of the base of the posterolateral processes of the parietals in the South African erythrosuchid Erythrosuchus africanus. A pair of tuberosities of similar shape in an equivalent position to those of Erythrosuchus africanus is also present in Garjainia madiba (Gower et al., 2014), Asperoris mnyama (Nesbitt, Butler & Gower, 2013), Batrachotomus kupferzellensis (SMNS 52970), and Prestosuchus chiniquensis (UFRGS-PV-0156-T). In other diapsids sampled here, the posterior surface of the base of the posteroventral process of the parietal is smooth.

171. Postparietal, size (pair of postparietals if they are not fused to each other): sheet-like, not much narrower than the suproccipital (0); small, splint-like (1); absent as a separate ossification (2) (Laurin, 1991: E2, G5, J2; reworded from Reisz, Laurin & Marjanović, 2010: 54; Ezcurra, Lecuona & Martinelli, 2010: 12, in part; Nesbitt, 2011: 146, in part; Dilkes & Arcucci, 2012: 17, in part; Ezcurra, Scheyer & Butler, 2014: 53; Pritchard et al., 2015: 24), ORDERED (Fig. 23).

See comments in Nesbitt (2011: character 146).

172. Postparietal, fusion between counterparts: absent (0); present, forming an interparietal (1) (Gauthier, 1984; Benton, 1985; Pritchard et al., 2015: 25). This character is inapplicable in taxa that lack postparietals.

The postparietals are paired plate-like bones that contact each other along the midline of the posterior end of the skull roof in non-saurian diapsids, including Petrolacosaurus kansensis (Reisz, 1981) and Youngina capensis (Gow, 1975). By contrast, the postparietal is represented by a single bone placed on the midline of the skull roof, immediately posterior to the paired parietals, in all the the saurians that possess this bone (e.g., Proterosuchus fergusi: RC 846, SAM-PK-K10603; Tasmaniosaurus triassicus: UTGD 54655; Erythrosuchus africanus: NMQR 1473; Euparkeria capensis: SAM-PK-5867).

173. Tabular: present (0); absent (1) (Laurin, 1991: E3; Reisz & Dilkes, 2003: 46; Reisz, Laurin & Marjanović, 2010: 53; Ezcurra, Scheyer & Butler, 2014: 52; Pritchard et al., 2015: 37).

The tabulars are a pair of bones exposed on the occipital surface of the skull and they do not contact with each other on the median line. The tabulars articulate on the posterior surface of the lateral end of the postparietals (e.g., Petrolacosaurus kansensis: Reisz, 1981) or the distal end of the posterolateral processes of the parietals (e.g., Youngina capensis: Gow, 1975). These bones are absent in saurians.

174. Palpebral/s: absent (0); present (1) (Nesbitt, 2011: 147) (Fig. 18).

See comments in Nesbitt (2011: character 147).

175. Neomorphic bone (=septomaxilla of phytosaurs), separate ossification anterior to nasals and surrounded by the premaxilla on the dorsal surface of the snout: absent (0); present (1) (Sereno, 1991; reworded from Nesbitt, 2011: 150) (Fig. 16 of Nesbitt, 2011).

See comments in Nesbitt (2011: character 150).

176. Quadrate, shape: straight posteriorly (0); shallowly emarginated (1); with conch (2) (Laurin, 1991: E7, J3; Müller, 2004: 29 in part; Reisz, Laurin & Marjanović, 2010: 55; Ezcurra, Scheyer & Butler, 2014: 54; Pritchard et al., 2015: 41, in part), ORDERED (Figs. 17 and 24).

The posterior margin of the quadrate is straight in lateral view in the non-saurian diapsids Petrolacosaurus kansensis (Reisz, 1981), Youngina capensis (BP/1/375, TM 3603), and Acerosodontosaurus piveteaui (MNHN 1908-32-57). By contrast, the quadrate is posteriorly bowed in saurians in lateral view. In particular, the anterior margin of the lateral surface of the quadrate is strongly laterally flared, forming a wing-like projection and a strongly concave lateral surface of the bone in lepidosaurormophs (e.g., Planocephalosaurus robinsonae: NHMUK PV R9967). This condition has been called quadrate conch and has historically been considered a synapomorphy of Lepidosauromorpha (Gauthier, Kluge & Rowe, 1988). In archosauromorphs, the lateral surface of the quadrate is only gently concave anteroposteriorly.

177. Quadrate, angle between the posterior margins of the dorsal and ventral ends: 41–47°(0); 91–96°(1); 106–137°(2); 143–158°(3) (New character), ORDERED. Inapplicable in taxa with a straight posterior margin of the quadrate (Figs. 17, 18 and 24).

This character describes the degree of curvature of the quadrate along its posterior margin in lateral view.

178. Quadrate, dorsal head: does not have a sutural contact with the paroccipital process of the opisthotic (0); has a sutural contact with the paroccipital process of the opisthotic (1) (Nesbitt, 2011: 77).

See comments in Nesbitt (2011: character 77).

179. Quadrate, dorsal head: partially exposed laterally (0); completely covered by the squamosal (1) (Sereno & Novas, 1994; Nesbitt, 2011: 78). This character is inapplicable in taxa that the quadrate is completely covered by the squamosal in lateral view (Figs. 17–19 of Nesbitt, 2011).

See comments in Nesbitt (2011: character 78).

180. Quadrate, dorsal end hooked posteriorly in lateral view: absent (0); present (1) (Nesbitt et al., 2015: 207) (Figs. 17 and 24).

See comments in Nesbitt et al. (2015: character 207).

181. Quadrate, foramen on the medial wall of the quadrate foramen: absent (0); present (1) (New character) (Fig. 24).

The medial wall of the quadrate foramen is smooth in the vast majority of diapsids, but in Pamelaria dolichotrachela (ISI R316/1), Sarmatosuchus otschevi (Gower & Sennikov, 1997), and Erythrosuchus africanus (Gower, 2003) the medial wall of the quadrate foramen is pierced by a laterally opening, subcircular foramen.

182. Quadrate, posterior margin of the ventral half in lateral view: concave (0); convex (1) (Nesbitt et al., 2015: 242). This character is inapplicable in taxa that possess a posteriorly straight quadrate (Fig. 24).

See comments in Nesbitt et al. (2015: character 242).

183. Quadrate, ventral condyles: subequally distally extended (0); medial condyle distinctly more distally projected than the lateral one (1) (New character).

This character describes the asymmetry in the ventral development of the ventral condyles of the quadrate. These condyles are subequally projected in most diapsids, but in some species the medial condyle is considerably more ventrally projected than the lateral one (e.g., Pamelaria dolichotrachela: ISI R316/1; Azendohsaurus madagaskarensis: UA-7-20-99-653).

184. Neomorph ossification, present between the pterygoid, quadrate, and skull roof: absent, the quadrate flange of the pterygoid meets the quadrate but remains free of the skull roof (0); present (1) (Evans, 1990: 12).

Choristoderans differ from other diapsids in the presence of a neomorphic bone that articulates with the pterygoid, quadrate, and skull roof (Sigogneau-Russell & Russell, 1978; Evans, 1990). This bone is plate-like and exposed in occipital view.

185. Vomer, shape: broad, plate-like bone, at least as transversely broad as the choanas (0); stick-like bone, transversely narrower than the choanas (1) (New character).

The vomer is a transversely broad bone that widely contributes to the anterior end of the palate and is approximately as broad as the choana (=internal naris) in non-archosauromorph diapsids (e.g., Petrolacosaurus kansensis: Reisz, 1981; Gephyrosaurus bridensis: Evans, 1980), choristoderans (e.g., Simoedosaurus lemoinei: Sigogneau-Russell & Russell, 1978), tanystropheids (e.g., Macrocnemus bassanii: Peyer, 1937; Tanystropheus longobardicus: Wild, 1973), allokotosaurians (e.g., Pamelaria dolichotrachela: ISI R316/1; Azendohsaurus madagaskarensis: Flynn et al., 2010) and rhynchosaurs (e.g., Mesosuchus browni: SAM-PK-6536). By contrast, the vomer is transversely narrow, being narrower than the choana immediately posterior to it, and forms a stick-like longitudinal bar with its counterpart on the anterior end of the palate in Prolacerta broomi (BP/1/2675) and archosauriforms (e.g., Proterosuchus fergusi: TM 201; “Chasmatosaurus” yuani: IVPP V36315; Chanaresuchus bonapartei: PULR 07; Parasuchus angustifrons: BSPG 1931 X 502).

186. Vomer, contact with maxilla: absent (0); present (1) (Dilkes, 1998: 38; Müller, 2004: 92; Ezcurra, Scheyer & Butler, 2014: 135; Pritchard et al., 2015: 46).

The vomer contacts the maxilla in choristoderans (Cteniogenys sp.: Evans, 1990; Simoedosaurus lemoinei: Sigogneau-Russell & Russell, 1978), rhynchosaurs (e.g., Mesosuchus browni: Dilkes, 1998; Bentonyx sidensis: BRSUG 21200), Proterosuchus goweri (NMQR 880), phytosaurs (Parasuchus agustifrons: BSPG 1931 X 502), ornithosuchids (e.g., Ornithosuchus longidens: Walker, 1964), and suchians (e.g., Batrachotomus kupferzellensis: Gower, 1999).

187. Vomer, teeth: present, more than one row or no rows are distinguishable (0); present, mainly in a single row, but multiple teeth present immediately anterior to the contact with the pterygoid (1); present, single row along entire extension (2); absent (3) (Dilkes, 1998; modified from Ezcurra, Lecuona & Martinelli, 2010: 37; Dilkes & Arcucci, 2012: 37, in part; Pritchard et al., 2015: 45, in part), ORDERED.

188. Palatine-pterygoid, teeth on the palatine and ventral surface of the anterior ramus of the pterygoid: present (0); absent (1) (Benton, 1985; modified from Pritchard et al., 2015: 44, 47) (Figs. 13, 24 and 26).

The presence of teeth on the ventral surface of the palatine and pterygoid is an ancestral condition of diapsids (Müller, 2004). However, the palatal dentition is completely lost in some archosauromorph taxa, such as Trilophosaurus buettneri (Spielmann et al., 2008), some rhynchosaurids (e.g., Bentonyx sidensis: BRSUG 21200), Prolacertoides jimusarensis (IVPP V3233), erythrosuchids (Gower, 2003), phytosaurs (e.g., Parasuchus angustifrons: BSPG 1931 X 502), ornithosuchids (e.g., Ornithosuchus longidens: Walker, 1964), most suchians (with the exception of some gracilisuchids; Butler et al., 2014a), and most ornithodirans (with the exception of Lewisuchus admixtus and some saurischians: Romer, 1972d; Sereno et al., 1993; Sereno, Martínez & Alcober, 2013; Bittencourt et al., 2014). In the tanystropheid Tanystropheus longobardicus the presence and absence of palatal teeth is ontogenetically related (Wild, 1973) and, as a result, this character is scored as polymorphic for this species.

189. Palatine-pterygoid, height and dimetre of teeth on the palatine, ventral surface of the anterior ramus of the pterygoid and vomer: considerably smaller than those of the marginal dentition (0); similar to those of the marginal dentition (1) (New character). This character is inapplicable in taxa lacking palatine-pterygoid teeth (Figs. 25 and 26).

Figure 25 Reconstructed pterygoids of non-archosaurian neodiapsids.

(A) Youngina capensis (modified from Gow, 1975; BP/1/3859; GHG K 106); (B) Gephyrosaurus bridensis (redrawn from Evans, 1980); (C) Azendohsaurus madagaskarensis (redrawn from Flynn et al., 2000); (D) Prolacerta broomi (modified from Gow, 1975); (E) Doswellia kaltenbachi (modified from Dilkes & Sues, 2009; USNM 214823); and (F) Chanaresuchus bonapartei (modified from Romer, 1971a; PULR 07; PVL 4575, 4586). Numbers indicate character-states scored in the data matrix and anterior direction is towards the top of the figure. Abbreviations: T1–4, tooth fields T1–T4. Not to scale.

Figure 26 Skulls (A, B, D–H) and isolated pterygoid (C) of Permo-Triassic archosauromorphs in ventral view.

(A) Mesosuchus browni (SAM-PK-6536); (B) Prolacerta broomi (BP/1/5066); (C) Erythrosuchus africanus (NHMUK PV R3592); (D) Chanaresuchus bonapartei (PVL 4586); (E) Doswellia kaltenbachi (USNM 214823); (F) Gracilisuchus stipanicicorum (MCZ 4117); (G) Parasuchus angustifrons; and (H) Herrerasaurus ischigualastensis. Numbers indicate character-states scored in the data matrix and the arrows indicate anterior direction. Scale bars equal 1 cm in (A, D–F), 5 mm in (B), 5 cm in (C, G), and 2 cm in (H). (G) Modified from Butler et al. (2014b) and (H) modified from Sereno & Novas (1994).

The palatal dentition is composed of tiny teeth, considerably smaller than those that make up the marginal dentition, in most diapsids. In some tanystropheids (e.g., Macrocnemus bassanii: BSPG 1973 I 86, cast of Bessano II; Tanystropheus longobardicus: Wild, 1973), some allokotosaurians (e.g., Pamelaria dolichotrachela: ISI R316/1; Azendohsaurus madagaskarensis: Flynn et al., 2010), and the rhynchosaurid Eohyosaurus wolvaardti (SAM-PK-K10159) the palatal teeth are approximately the same height and dimetre at the base of the crown as those of the marginal dentition.

190. Palatine, transverse extension: narrow, subequal contribution of the palatine and pterygoid to or pterygoid main component of the palate posteriorly to the choanae (0); broad, the palatine is the main component of the palate posteriorly to the choanae and the anterior ramus of the pterygoid is splint-like (1) (taken from Liu et al., 2015; new character for quantitative phylogenies). This character can be scored also from the shape of the anterior ramus of the pterygoid (Fig. 26).

Figure 27 Braincases of Triassic archosauromorphs in occipital view.

(A) Azendohsaurus madagaskarensis (UA 7-20-99653); (B) Proterosuchus goweri (NMQR 880); (C) Euparkeria capensis (UMZC T692); (D) Chanaresuchus bonapartei (MCZ 4037); (E) Doswellia kaltenbachi (USNM 214823); (F) Parasuchus hislopi (ISI R42); and (G) Batrachotomus kupferzellensis (SMNS 80260). Numbers indicate character-states scored in the data matrix. Scale bars equal 1 cm in (A, B, F, G), 2 mm in (C), and 5 mm in (D, E).

In most non-archosaurian archosauromorphs, with the exception of allokotosaurians (e.g., Azendohsaurus madagaskarensis: Flynn et al., 2010; Trilophosaurus buettneri: Spielmann et al., 2008) and Euparkeria capensis (Ewer, 1965), the anterior ramus of the pterygoid is transversely broad and it is the main component or contributes about as much as the palatine to the palate posterior to the chonae. A similar condition is also present in the lepidosauromorphs sampled here (Evans, 1980; Fraser, 1982). By contrast, the palatine is transversely broad and represents the main component of the palate posterior to the chonae in phytosaurs (e.g., Parasuchus angustifrons: BSPG 1931 X 502), ornithosuchids (e.g., Ornithosuchus longidens: Walker, 1964; Riojasuchus tenuisceps: PVL 3827), suchians (e.g., Turfanosuchus dabanensis: IVPP V3237), dinosaurs (e.g., Herrerasaurus ischigualastensis: PVSJ 407), and also the archosaur “Chasmatosaurus ultimus” (IVPP V2301).

191. Palatine, anterior processes forming the posterior border of the choana: subequal in anterior extension or anterolateral process longer (0); anteromedial process longer (1); single process (2) (New character) (Fig. 26).

In the vast majority of diapsids, the posterior border of the choana is formed by a pair of anteriorly projecting processes, one anteromedial and one anterolateral, of the palatine. Only in Petrolacosaurus kansensis (Reisz, 1981) and Chanaresuchus bonapartei (PULR 07) are these processes subequal in length or the anterolateral process is the longest. Otherwise, the anteromedial process is the longest. Phytosaurs possess a unique condition among archosauromorphs, in which the palatine has a single anterior process (e.g., Parasuchus angustifrons: BSPG 1931 X 502; Parasuchus hislopi: ISI R43; Nicrosaurus kapffi: NHMUK PV R42743; Smilosuchus gregorii: Camp, 1930).

192. Pterygoids, contact with each other: present, anteriorly (0); absent, remain separate along their entire length (1) (Dilkes, 1998: 126; Ezcurra, Lecuona & Martinelli, 2010: 41; Ezcurra, Scheyer & Butler, 2014: 137; Pritchard et al., 2015: 52, in part) (Fig. 26).

This character describes the absence of a contact between the anterior ramus of the pterygoid and its counterpart at any point along the midline of the palate. The condition of this character is usually only unambiguously determined if the pterygoids are preserved in natural articulation in the palate and they are undistorted or only slightly distorted. On the other hand, the contact between the pterygoids is difficult to determine if the palate is distorted or partially disarticulated or if the pterygoid is isolated.

193. Pterygoid, anterior ramus (=palatal process): extends anterior to the anterior limit of the palatine (0); forms oblique suture with palatine but process ends before reaching anterior limit of palatine (1); forms transverse suture with palatine (2) (DeBraga & Rieppel, 1997; Müller, 2004: 139; Ezcurra, Scheyer & Butler, 2014: 138) (Fig. 26).

The anterior ramus of the pterygoid extends anteriorly to the most anterior tip of the palatine in most diapsids, but the inverse condition is present in the following taxa sampled here: early rhynchocephalians (e.g., Gephyrosaurus bridensis: Evans, 1980), choristoderans (e.g., Simoedosaurus lemoinei: Sigogneau-Russell & Russell, 1978), Euparkeria capensis (Ewer, 1965), proterochampsids (e.g., Proterochampsa barrionuevoi: Dilkes & Arcucci, 2012), phytosaurs (e.g., Parasuchus angustifrons: BSPG 1931 X 502; Nicrosaurus kapffi: NHMUK PV R42743), and dinosaurs (e.g., Heterodontosaurus tucki: Norman et al., 2011).

194. Pterygoid, anterior ramus (=palatal process) shape: transversely broad at its base, converging gradually with the transverse ramus (0); transversely narrow along its entire extension, converging in a right or acute angle with the transverse ramus, with the bone possessing an overall L-shape contour in ventral or dorsal view (1) (New character) (Fig. 26).

The anterior ramus of the pterygoid is transversely broad at its base and tapers gradually anteriorly in ventral or dorsal view in most diapsids. However, the anterior ramus of the pteryoid is transversely narrow, restricted to a splint-like process, in Trilophosaurus buettneri (Spielmann et al., 2008), phytosaurs (e.g., Parasuchus angustifrons: BSPG 1931 X 502; Nicrosaurus kapffi: NHMUK PV R42743), ornithosuchids (e.g., Ornithosuchus longidens: NHMUK PV R3143; Riojasuchus tenuisceps: PVL 3827), suchians (e.g., Turfanosuchus dabanensis: IVPP V3237; Gracilisuchus stipanicicorum: MCZ 4117; Aetosauroides scagliai: PVL 2059), and the archosaur “Chasmatosaurus ultimus” (IVPP V2301). The pterygoid acquires an L-shaped morphology in ventral or dorsal view in these taxa.

195. Pterygoid, teeth on the ventral surface of the anterior ramus (=palatal process), excluding tiny palatal teeth if present: present in two distinct fields (=T2 and T3 of Welman, 1998) (0); present in three distinct fields (=T2, T3a and T3b) (1); present in three distinct fields (=T2a, T2b and T3) (2); present in one field that occupies most of the transverse width of the ramus (=T2 + T3) (3); present in only one posteromedially-to-anterolaterally oriented field (=T2) (4); present in only one field adjacent to the medial margin of the ramus (=T3) (5) (Dilkes, 1998: 68; Müller, 2004: 100; Ezcurra, Lecuona & Martinelli, 2010: 39, in part; Nesbitt, 2011: 175, in part; modified from Dilkes & Arcucci, 2012: 39; modified from Ezcurra, Scheyer & Butler, 2014: 139; Pritchard et al., 2015: 48, 49, in part). This character is inapplicable in taxa that lack teeth in the palatine and the ventral surface of the anterior ramus of the pterygoid (Figs. 25 and 26).

The disposition of the palatal teeth on the anterior ramus of the pterygoid has been used by many previous authors as a character in quantitative phylogenetic analyses focused on diapsid or, more specifically, early archosauromorph interrelationships (e.g., Dilkes, 1998; Müller, 2004; Ezcurra, Lecuona & Martinelli, 2010; Nesbitt, 2011; Dilkes & Arcucci, 2012; Ezcurra, Scheyer & Butler, 2014; Pritchard et al., 2015). However, the states of these characters were restricted to the mere presence of palatal teeth or the number of fields of palatal teeth on the anterior ramus of the pterygoid, but they did not consider alternative positions of these fields. Here, five different states are used for this character and the nomenclature of Welman (1998) is employed to describe the position of the different fields. Welman (1998) described two fields on the ventral surface of the anterior ramus of the pterygoid, namely T2 and T3. T2 is placed adjacent to the lateral margin of the anterior ramus, and it is usually anterolaterally-to-posteromedially oriented and continues onto the palatine. T3 is adjacent to the medial margin of the anterior ramus and may extend immediately anterior to the basal articulation and continue anteriorly onto the vomer. T4 is also present on the anterior ramus of the pterygoid, but since it is restricted to the ventromedial margin of the process it is here considered an independent character. The presence of tiny palatal teeth, considerably smaller than others (e.g., Petrolacosaurus kansensis: Reisz, 1981), was not considered in determining the distribution of fields.

The palatal teeth on the anterior ramus of the pterygoid are distributed in three fields as a result of a subdivision of T3 (here labelled as T3a and T3b) in the rhynchocephalians Gephyrosaurus bridensis (Evans, 1980) and Planocephalosaurus robinsonae (Fraser, 1982) and in the early allokotosaurian Pamelaria dolichotrachela (ISI R316/1). Three fields of palatal teeth are also present in Azendohsaurus madagaskarensis (Flynn et al., 2010), but this is a result of the subdivision of T2 (here labelled as T2a and T2b), contrasting with the latter species. The most broadly distributed condition among diapsids is the presence of two fields (T2 and T3), which is observed in Petrolacosaurus kansensis (Reisz, 1981), Youngina capensis (BP/1/3859), Macrocnemus bassanii (Peyer, 1937), early rhynchosaurs (e.g., Mesosuchus browni: SAM-PK-6536; Howesia browni: Dilkes, 1995), Prolacerta broomi (BP/1/2675), proterosuchids (e.g., Proterosuchus fergusi: RC 59; Proterosuchus goweri: NMQR 880; Proterosuchus alexanderi: NMQR 1484), Sarmatosuchus otschevi (PIN 2865/68), the holotype of Garjainia prima (PIN 2394/5), Euparkeria capensis (Ewer, 1965), proterochampsids (e.g., Tropidosuchus romeri: PVL 4601; Gualosuchus reigi: PULR 05; Chanaresuchus bonapartei: PULR 07; Pseudochampsa ischigualastensis: Trotteyn & Ezcurra, 2014), and Doswellia kaltenbachi (USNM 214823). In some diapsids, most of the ventral surface of the anterior ramus of the pterygoid possesses teeth that cannot be differentiated in fields, which is the condition in Simoedosaurus lemoinei (Sigogneau-Russell & Russell, 1978), Amotosaurus rotfeldensis (SMNS 50691), Nundasuchus songeaensi (Nesbitt et al., 2014), and Turfanosuchus dabanensis (IVPP V3237). Finally, a single field of palatal teeth, identified as T3, is present in the dinosauriform Lewisuchus admixtus (PULR 01) and juveniles of Tanystropheus longobardicus (Wild, 1973; Nosotti, 2007), and a single field identified as T2 is present in Proterochampsa barrionuevoi (Dilkes & Arcucci, 2012).

196. Pterygoid, number of rows on palatal tooth field T2: more than two or do not dispose on distinct rows (0); two rows parallel to each other (1); single row (2) (New character). This character is inapplicable if the tooth field T2 is subdivided in T2a and T2b or is absent (Figs. 25 and 26).

Among diapsids that possess a field T2 (sensu Welman, 1998) of palatal teeth on the anterior ramus of the pterygoid, there are more than two rows or no distinct rows in this field in Petrolacosaurus kansensis (Reisz, 1981), Youngina capensis (BP/1/3859), Mesosuchus browni (SAM-PK-6536), Prolacerta broomi (BP/1/5066), proterosuchids (e.g., Proterosuchus fergusi: RC 59; Proterosuchus alexanderi: NMQR 1484) and Doswellia kaltenbachi (USNM 214823). By contrast, there are two distinct rows of teeth in the field T2 in Gephyrosaurus bridensis (Evans, 1980), Cteniogenys sp. (Evans, 1990), Macrocnemus bassanii (Peyer, 1937) and Howesia browni (Dilkes, 1995), whereas there is a single row in Planocephalosaurus robinsonae (NHMUK PV R9960; Fraser, 1982), the holotype of Garjainia prima (PIN 2394/5), Euparkeria capensis (Ewer, 1965), and proterochampsids (e.g., Tropidosuchus romeri: PVL 4601; Chanaresuchus bonapartei: PULR 07; Pseudochampsa ischigualastensis: Trotteyn & Ezcurra, 2014; Proterochampsa barrionuevoi: Dilkes & Arcucci, 2012).

197. Pterygoid, number of rows on palatal tooth field T3: more than two or not disposed in distinct rows (0); two parallel rows (1); single row (2) (New character). This character is inapplicable if the tooth field T3 is subdivided into T3a and T3b or is absent (Figs. 25 and 26).

This character describes the same states of the character 196, but for the field of palatal teeth T3. In most of the diapsids sampled here, T3 possesses more than two or do not present distinct rows of teeth (e.g., Petrolacosaurus kansensis: Reisz, 1981; Cteniogenys sp.: Evans, 1990), early rhynchosaurs (e.g., Mesosuchus browni: SAM-PK-6536; Howesia browni: Dilkes, 1995), Prolacerta broomi (BP/1/5066),proterosuchids (e.g., Proterosuchus fergusi: RC 59; Proterosuchus alexanderi: NMQR 1484; “Chasmatosaurus” yuani: IVPP V36315), Sarmatosuchus otschevi (PIN 2865/68), and Euparkeria capensis (Ewer, 1965). The field T3 is formed by two parallel rows of teeth in Youngina capensis (Gow, 1975) and Azendohsaurus madagaskarensis (Flynn et al., 2010), but there is a single row in Macrocnemus bassanii (Peyer, 1937), Tanystropheus longobardicus (Wild, 1973), the holotype of Garjainia prima (PIN 2394/5), proterochampsids (e.g., Tropidosuchus romeri: PVL 4601; Chanaresuchus bonapartei: PULR 07; Pseudochampsa ischigualastensis: Trotteyn & Ezcurra, 2014), Doswellia kaltenbachi (USNM 214823), and Lewisuchus admixtus (PULR 01).

198. Pterygoid, most lateral row of teeth on the ventral surface of the anterior ramus (tooth field T2) raised on a thick, posteromedially-to-anterolaterally oriented ridge: absent (0); present (1) (New character). This character is inapplicable if tooth field T2 is absent (Figs. 25 and 26).

Proterochampsids (e.g., Proterochampsa barrionuvoi: PVL 2063; Pseudochampsa ischigualastensis: Trotteyn & Ezcurra, 2014; Chanaresuchus bonapartei: PULR 07; Tropidosuchus romeri: PVL 4601) and the doswelliid Doswellia kaltenbachi (USNM 214823) possess a thick and prominent anterolaterally-to-posteromedially oriented ridge that bears field T2 of palatal teeth on the pterygoid. By contrast, the lateral row of palatal teeth is not elevated from the rest of the surface of the pterygoid in other diapsids.

199. Pterygoid, a row of fang-like teeth on the medial edge of the anterior ramus (=palatal process) (=T4 of Welman, 1998): absent (0); present (1) (New character) (Figs. 25 and 26).

The palatal dentition on the pterygoid is restricted to the ventral surface of the bone in almost all diapsids. As a result, the presence of a single row of fang-like teeth on the medial edge of the anterior ramus of the pterygoid (=T4 of Welman, 1998) in Prolacerta broomi (BP/1/2675; Gow, 1975), Boreopricea funerea (Tatarinov, 1978: 510), Proterosuchus fergusi (RC 59), Proterosuchus goweri (NMQR 880), Proterosuchus alexanderi (NMQR 1484), and Tasmaniosaurus triassicus (UTGD 54655; Ezcurra, 2014) represents an unusual condition among diapsids.

200. Pterygoid, orientation of the lateral ramus: posterolaterally, forming an obtuse angle with the anterior ramus (0); laterally or anterolaterally, forming a right or acute angle with the anterior ramus (1) (New character) (Figs. 13, 25 and 26).

The main axis of the posterior margin of the lateral ramus of the pterygoid is posterolaterally oriented in ventral or dorsal view in most species historically grouped within “Proterosuchia”, including Proterosuchus fergusi (RC 59; SAM-PK-11208), Proterosuchus alexanderi (NMQR 1484), “Chasmatosaurus” yuani (IVPP V4067), Sarmatosuchus otschevi (PIN 2865/68), Guchengosuchus shiguaiensis (IVPP V8808), Garjainia prima (PIN 2394/5), and Erythrosuchus africanus (NHMUK PV R3592). By contrast, the lateral ramus of the pterygoid is directly laterally oriented in the erythrosuchid Shansisuchus shansisuchus (Young, 1964) and most other diapsids sampled here.

201. Pterygoid, lateral margin of the lateral ramus in dorsal or ventral view: posterolateral margin with an acute corner (0); posterolateral margin merges smoothly into anterolateral margin forming a smoothly convex lateral outline (1) (DeBraga & Rieppel, 1997; Müller, 2004: 164; Ezcurra, Scheyer & Butler, 2014: 141) (Figs. 25 and 26).

202. Pterygoid, teeth on the lateral ramus: present, more than a single row or no rows recognizable (0); present, single row on the posterior edge (=T1 of Welman, 1998) (1); absent (2) (Laurin, 1991: E5; Reisz & Dilkes, 2003: 30; Müller, 2004: 163; Reisz, Laurin & Marjanović, 2010: 49; Ezcurra, Lecuona & Martinelli, 2010: 40, in part; Nesbitt, 2011: 176, in part; Dilkes & Arcucci, 2012: 40, in part; modified from Ezcurra, Scheyer & Butler, 2014: 48; Pritchard et al., 2015: 50, 51), ORDERED (Figs. 13, 25 and 26).

See comments in Nesbitt (2011: character 176).

203. Ectopterygoid, body: arcs anteriorly (0); arcs anterodorsally (1) (Nesbitt, 2011: 87).

See comments in Nesbitt (2011: character 87) (Fig. 22 of Nesbitt, 2011).

204. Ectopterygoid, articulation with pterygoid: simple overlap between ectopterygoid and pterygoid (0); interlaced, complex overlap between ectopterygoid and pterygoid (1) (Sereno & Novas, 1994; Dilkes, 1998: 142; Nesbitt, 2011: 84; Ezcurra, Scheyer & Butler, 2014: 142) (Fig. 26).

See comments in Nesbitt (2011: character 84).

205. Ectopterygoid, suture with pterygoid: does not reach the posterolateral corner of the transverse flange (0); reaches the posterolateral corner of the transverse flange (1) (Dilkes, 1998: 42; Müller, 2004: 95; Ezcurra, Lecuona & Martinelli, 2010: 158; Nesbitt, 2011: 88, in part; modified from Ezcurra, Scheyer & Butler, 2014: 143, and Nesbitt et al., 2015: 206) (Fig. 26).

See comments in Nesbitt (2011: character 88).

206. Ectopterygoid, contact with maxilla: absent (0); present (1) (Dilkes, 1998: 40; Müller, 2004: 94; Ezcurra, Lecuona & Martinelli, 2010: 19; Dilkes & Arcucci, 2012: 27; Ezcurra, Scheyer & Butler, 2014: 144) (Fig. 26).

The anterior end of the lateral projection of the ectopterygoid contacts the distal tip of the maxilla and continues posteriorly to its contact with the lateral wall of the skull on the medial surface of the jugal in a wide range of diapsids (e.g., Gephyrosaurus bridensis: (Evans, 1980); Simoedosaurus lemoinei: Sigogneau-Russell & Russell, 1978; Trilophosaurus buettneri: Spielmann et al., 2008; Rhynchosaurus articeps: NHMUK PV R1236; Garjainia prima: PIN 2394/5; Chanaresuchus bonapartei: PULR 07; Doswellia kaltenbachi: USNM 214823; Parasuchus angustifrons: BSPG 1931 X 502). By contrast, the ectoperygoid is restricted to a contact with the jugal in other species. The presence of an ectopterygoid-maxilla contact does not seem to show a clear phylogenetic signal based on the phylogenetic relationships of archosauromorphs found by previous authors (e.g., Dilkes, 1998; Nesbitt, 2011), but it may be informative at low taxonomic levels.

207. Ectopterygoid, posterior expansion in contact with jugal: absent (0); present (1) (Dilkes, 1998: 39; Ezcurra, Scheyer & Butler, 2014: 145) (Figs. 7 and 26).

The lateral process of the quadratojugal is subrectangular or trapezoidal, with a slight anteroposterior expansion of its distal end, in non-archosauromorph diapsids, tanystropheids (e.g., Tanystropheus longobardicus: Wild, 1973), Azendohsaurus madagaskarensis (Flynn et al., 2010), and Prolacertoides jimusarensis (IVPP V3233). By contrast, the lateral process of the quadratojugal is posteriorly expanded, acquiring a hook-shape in dorsal or ventral view, in rhynchosaurs (e.g., Mesosuchus browni: SAM-PK-6536; Howesia browni: Dilkes, 1995) and archosauriforms (Dilkes, 1998).

208. Supraoccipital, shape in occipital view: plate-like (0); inverted V-shape (1) (Dilkes, 1995; Pritchard et al., 2015: 56).

The supraoccipital is usually a pentagonal bone, with a horizontal ventral margin and a dorsally oriented apex in most diapsids. By contrast, the supraoccipital has straight lateral margins that converge in a dorsal apex and a notched ventral margin in the choristoderan Simoedosaurus lemoinei (MNHN.F.BL9022) and rhynchosaurids (e.g., Rhynchosaurus articeps: Benton, 1990; Bentonyx sidensis: Hone & Benton, 2008).

209. Supraoccipital, participation in the dorsal border of the foramen magnum: absent (0); present (1) (Gower, 2002; Ezcurra, Lecuona & Martinelli, 2010: 152; Nesbitt, 2011: 126; Pritchard et al., 2015: 60) (Fig. 27).

See comments in Nesbitt (2011: character 126).

210. Supraoccipital, posterior surface: smooth or with a low median ridge (0); with a prominent median, vertical ridge (1) (Dilkes & Sues, 2009; Desojo, Ezcurra & Schultz, 2011: 94; Pritchard et al., 2015: 55) (Fig. 27).

211. Otoccipital, fusion between opisthotic and exoccipital: absent or partial (0); present (1) (Juul, 1994; Ezcurra, Lecuona & Martinelli, 2010: 22; Dilkes & Arcucci, 2012: 29; Pritchard et al., 2015: 62) (Fig. 27).

212. Opisthotic, contact between paroccipital process and parietal immediately lateral to supraoccipital: absent (0); present (1) (Dilkes, 1998; Ezcurra, Lecuona & Martinelli, 2010: 31; Dilkes & Arcucci, 2012: 36) (Fig. 27).

213. Opisthotic, paroccipital processes orientation: extend laterally forming aproximately a 90°angle with the parasagittal plane (0); deflected posterolaterally at an angle of more than 20°from the transverse plane of the skull (1) (DeBraga & Rieppel, 1997; Müller, 2004: 158; modified from Ezcurra, Scheyer & Butler, 2014: 146) (Fig. 16).

214. Opisthotic, paroccipital process attachment: ends freely (0); contacts supratemporal or proximal end of quadrate and/or squamosal (1); sutured to the pterygoid and the pterygoid wing of the quadrate (2) (modified from Evans, 1990: 14; Laurin, 1991: A4, E6; Reisz & Dilkes, 2003: 26; Reisz, Laurin & Marjanović, 2010: 58; modified from Ezcurra, Scheyer & Butler, 2014: 57; Pritchard et al., 2015: 58).

215. Opisthotic, paroccipital process morphology: unflattened and tapered (0); anteroposteriorly-flattened distally (1) (Gauthier, 1984; Gauthier, Kluge & Rowe, 1988; Pritchard et al., 2015: 63) (Fig. 30).

The paroccipital process of the opisthotic is gracile, with an anteroposteriorly thin distal end, in Azendohsaurus madagaskarensis (UA 7-20-99-653), rhynchosaurs (e.g., Mesosuchus browni: SAM-PK-6536; Bentonyx sidensis: BRSUG 21200), Prolacerta broomi (BP/1/2675), Kadimakara australiensis (QM F6710), and archosauriforms (e.g., Proterosuchus alexanderi: NMQR 1484; Sarmatosuchus otschevi: PIN 2865/68-1; Garjainia prima: PIN 2394/5; Euparkeria capensis: SAM-PK-5867; Chanaresuchus bonapartei: PULR 07; Doswellia kaltenbachi: USNM 214823; Parasuchus angustifrons: BSPG 1931 X 502; Turfanosuchus dabanensis: IVPP V3237; Lewisuchus admixtus: PULR 01). Conversely, the paroccipital process tapers continuously towards its distal end in dorsal or ventral view and lacks the narrow distal end in non-archosauromorph diapsids and tanystropheids (e.g., Protorosaurus speneri: BSPG 1995 I 5, cast of WMsN P47361; Tanystropheus longobardicus: Wild, 1973).

216. Opisthotic, fossa immediately lateral to the foramen magnum: absent (0); present (1) (taken from Gower & Sennikov, 1996; new character for quantitative phylogeny).

Gower & Sennikov (1996) described the presence of a pair of oval fossae immediately lateral to the foramen magnum on the posterior surfaces of the opisthotics of Fugusuchus hejipanensis and Garjainia prima. A very similar condition is also present in the early rhynchosaurs Mesosuchus browni (SAM-PK-6536) and Howesia browni (Dilkes, 1995) and in the erythrosuchid Garjainia madiba (BP/1/5760). This pair of fossae seems to not be related to cranial nerve foramina and a soft-tissue correlate is unknown. The posterior surface of the opisthotic immediately lateral to the foramen magnum is flat in other diapsids sampled here.

217. Opisthotic, ventral ramus shape: club-shaped (0); pyramidal, with a tapering distal end (1); rod-like, with a cylindrical distal end and relatively thin (2); rod-like and very robust (3); plate-like (4) (Gower & Sennikov, 1996; Gower & Sennikov, 1997; Dilkes, 1998: 46; Ezcurra, Lecuona & Martinelli, 2010: 105, in part; modified from Ezcurra, Scheyer & Butler, 2014: 148; Pritchard et al., 2015: 57, in part) (Fig. 28).

Figure 28 Braincases of Triassic archosauromorphs in (A–F) ventral and (G–J) lateral views.

(A) Prolacerta broomi (BP/1/2675); (B) Proterosuchus alexanderi (NMQR 1484); (C, I) Garjainia madiba (BP/1/5525); (D) Doswellia kaltenbachi (USNM 214823); (E) Lewisuchus admixtus (PULR 01); (F, H) Parasuchus hislopi (ISI R42, mirrored); (G) Proterosuchus goweri (NMQR 880); and (J) Batrachotomus kupferzellensis (SMNS 80260). Numbers indicate character-states scored in the data matrix and the arrows indicate anterior direction. Scale bars equal 2 mm in (A), 1 cm in (B, C, F–J), and 5 mm in (D, E).

The morphology of the ventral ramus of the opisthotic has been widely used in previous phylogenetic analyses focused on archosauriform interrelationships (e.g., Gower & Sennikov, 1996; Gower & Sennikov, 1997; Dilkes, 1998; Ezcurra, Lecuona & Martinelli, 2010; Pritchard et al., 2015). Here, the shape of the ventral ramus of the opisthotic has been qualitatively discretized into four states. In Youngina capensis (Gardner, Holliday & O’Keefe, 2010), Simoedosaurus lemoinei (MNHN.F.BL9022), and Protorosaurus speneri (BSPG 1995 I 5, cast of WMsN P47361), the ventral ramus of the opisthotic is pyramidal, being broad at its base and tapering continuously towards its distal end. By contrast, the distal end of this process is transversely expanded and rounded in occipital view in Prolacerta broomi (BP/1/2675), Proterosuchus fergusi (BP/1/3993), Proterosuchus alexanderi (NMQR 1484), Fugusuchus hejiapanensis (Gower & Sennikov, 1996) and Sarmatosuchus otschevi (PIN 2865/68-1). Another distinct morphology is represented by a cylindrical ventral ramus, being non-expanded neither tapering, which is widely distributed among non-archosaurian archosauromorphs. In particular, a cylindrical and gracile process is present in Azendohsaurus madagaskarensis (UA 7-20-99-653), Mesosuchus browni (SAM-PK-6536), Chanaresuchus bonapartei (PULR 07), and Doswellia kaltenbachi (USNM 214823), but a considerably more robust cylindrical process that extends considerably laterally from the lateral surface of the basioccipital is present in Trilophosaurus buettneri (Spielmann et al., 2008), Proterosuchus goweri (NMQR 880), “Chasmatosaurus” yuani (IVPP V2719), Garjainia prima (Gower & Sennikov, 1996), and Garjainia madiba (Gower et al., 2014). Gower (1996) and Gower & Sennikov (1996) described the presence of a plate-like ventral ramus of the opisthotic for the erythrosuchids Erythrosuchus africanus and Shansisuchus shansisuchus, and a similar condition is also present in most more crownward archosauriforms, including Euparkeria capensis (Gower & Weber, 1998), Dorosuchus neoetus (PIN 1579/6-2), Proterochampsa barrionuevoi (Trotteyn & Haro, 2011), Rhadinosuchus gracilis (Ezcurra, Desojo & Rauhut, 2015), Archeopelta arborensis (Desojo, Ezcurra & Schultz, 2011), Parasuchus angustifrons (BSPG 1931 X 502), Turfanosuchus dabanensis (IVPP V3237), Batrachotomus kupferzellensis (SMNS 80260), Lewisuchus admixtus (PULR 01), and Herrerasaurus ischigualastensis (PVSJ 407).

218. Opisthotic, ventral ramus: extends further laterally than the lateralmost edge of the exoccipital in posterior view (0); covered by the lateralmost edge of the exoccipital in posterior view (1) (Gower & Sennikov, 1996; Gower, 2002; Nesbitt, 2011: 111) (Fig. 27).

See comments in Nesbitt (2011: character 111).

219. Exoccipital, morphology of the dorsal end: exoccipital columnar throught dorsoventral height, forming transversely narrow dorsal contact with more dorsal occipital elements (0); dorsal portion of exoccipital exhibits dorsomedially inclined process that forms transversely broad contact with more dorsal occipital elements (1) (Pritchard et al., 2015: 59). This character is inapplicable in taxa without a discernable opisthotic-exoccipital suture.

The morphology of the dorsal end of the exoccipital cannot be determined because of the fusion between this bone and the opisthotic in most archosauriforms. In the vast majority of non-archosauromorph diapsids sampled here (with the exception of Simoedosaurus lemoinei) and tanystropheids (e.g., Tanystropheus longobardicus: Wild, 1973), the dorsal end of the exoccipital is not expanded in posterior view and possesses transversely narrow contacts with other occipital bones. By contrast, the dorsal end of the exoccipital is medially expanded and possesses transversely broad sutures with other occipital bones in allokotosaurians (e.g., Pamelaria dolichotrachela: ISI R316/1; Azendohsaurus madagaskarensis: UA 7-20-99-653; Trilophosaurus buettneri: Spielmann et al., 2008), rhynchosaurs (e.g., Mesosuchus browni: SAM-PK-6536), Prolacerta broomi (BP/1/2675), and early archosauriforms (e.g., Proterosuchus fergusi: BP/1/3993; Proterosuchus goweri: NMQR 880).

220. Exoccipital, lateral surface: without subvertical crest (=metotic strut) (0); with clear crest (=metotic strut) present posterior to external foramina for hypoglossal nerve (CN XII) (1); with clear crest (=metotic strut) present anterior to the more posterior external foramina for hypoglossal nerve (CN XII) (2) (Gower, 2002; Nesbitt, 2011: 114) (Fig. 28).

See comments in Nesbitt (2011: character 114).

221. Exoccipital, medial margin of their distal ends: no contact with its counterpart (0); contact with its counterpart to exclude basioccipital from the floor of the endocranial cavity and diverge from each other on the occipital condyle, exposing the basioccipital dorsally (1); contact with its counterpart along the entire dorsal surface of the basioccipital, excluding the basioccipital from the floor of the endocranial cavity and the dorsal surface of the occipital condyle (2) (Gower & Sennikov, 1996; Ezcurra, Lecuona & Martinelli, 2010: 32; Nesbitt, 2011: 115; modified from Pritchard et al., 2015: 61), ORDERED (Fig. 27).

See comments in Gower & Sennikov (1996: character 17) and Nesbitt (2011: character 115).

222. Exoccipital, number of foramina for the passage of the hypoglossal nerve (CN XII): two (0); one (1) (Gower & Sennikov, 1996; Ezcurra, Lecuona & Martinelli, 2010: 23; Dilkes & Arcucci, 2012: 30).

See comments in Gower & Sennikov (1996: character 16).

223. Pseudolagenar recess, opening externally between the ventral surface of the ventral ramus of the opisthotic and the basal tubera: present (0); absent (1) (Gower & Sennikov, 1996; Gower & Sennikov, 1997; Ezcurra, Lecuona & Martinelli, 2010: 111) (Fig. 27).

See comments in Gower & Sennikov (1996: character 14).

224. Lagenar/cochlear recess: absent or short and strongly tapered (0); present and elongated and tubular (1) (Gower, 2002; Nesbitt, 2011: 118).

See comments in Nesbitt (2011: character 118).

225. Basioccipital-parasphenoid/parabasisphenoid, contact with each other in mature individuals: loose, overlapping suture (0); tightly sutured, sometimes by an interdigitated suture, or both bones fused to each other (1) (DeBraga & Rieppel, 1997; Müller, 2004: 137; modified from Ezcurra, Scheyer & Butler, 2014: 151) (Fig. 28).

The articulation between the basioccipital and the parabasisphenoid is tightly sutured, usually represented by an interdigitated suture, or both bones are fused to each other in mature individuals of Simoedosaurus lemoinei (MNHN.F.BL9022, BR10199), some non-archosauriform archosauromorphs (e.g., Azendohsaurus madagaskarensis: UA 7-20-99-653; Mesosuchus browni: SAM-PK-6536) and most archosauriforms to the exclusion of proterosuchids (e.g., Sarmatosuchus otschevi: PIN 2865/68-1; Proterochampsa barrionuevoi: PVL 2063; Doswellia kaltenbachi: USNM 214823). By contrast, the basioccipital and parasphenoid/parabasisphenoid articulate with each other along a loose suture in most non-archosauromorph diapsids (e.g., Petrolacosaurus kansensis: Reisz, 1981; Youngina capensis: Gardner, Holliday & O’Keefe, 2010; Gephyrosaurus bridensis: Evans, 1980), tanystropheids (e.g., Tanystropheus longobardicus: Wild, 1973), Pamelaria dolichotrachela (ISI R316/1), Prolacerta broomi (BP/1/2675), proterosuchids (e.g., Proterosuchus goweri: NMQR 880; Proterosuchus alexanderi: NMQR 1484; Proterosuchus fergusi: BP/1/3993), Vancleavea campi (Nesbitt et al., 2009), and Gracilisuchus stipanicicorum (MCZ 4117).

226. Basioccipital-parasphenoid/parabasisphenoid, basal tubera: absent (0); present (1) (DeBraga & Rieppel, 1997; Pritchard et al., 2015: 64) (Fig. 27).

The basal tubera are a pair of ventral or ventrolateral projections formed by both basioccipital and parasphenoid/parabasisphenoid. The basal tubera are broadly visible below the occipital condyle in occipital view. All the terminals sampled here possess these structures with the exception of the early diapsid Petrolacosaurus kansensis.

227. Basioccipital-parasphenoid/parabasisphenoid, basal tubera shape: clearly separated from each other (0); partially in contact with each other (1); medially expanded and nearly or completely connected (2) (modified from Nesbitt, 2011: 104) ORDERED. This character is inapplicable in taxa lacking basal tubera (Fig. 27).

See comments in Nesbitt (2011: character 104).

228. Basioccipital, position of the posterior margin of the occipital condyle: level with craniomandibular joint (0); anterior to craniomandibular joint (1); posterior to craniomandibular joint (2) (Dilkes, 1998: 51; Ezcurra, Lecuona & Martinelli, 2010: 27; Ezcurra, Scheyer & Butler, 2014: 157) (Fig. 26).

229. Basioccipital, articular surface of the occipital condyle: concave (0); hemispherical (1) (New character) (Fig. 28).

The occipital condyle of choristoderans (e.g., Cteniogenys sp.: Evans, 1990; Simoedosaurus lenoinei: MNHM.F.BL9022) and archosauromorphs (Ezcurra, et al., 2015) is subspherical, with a continuously convex facet for articulation with the postcranial axial skeleton. By contrast, the occipital condyle possesses a deeply concave posterior articular surface in Petrolacosaurus kansensis (Reisz, 1981), Youngina capensis (Gardner, Holliday & O’Keefe, 2010), and the lepidosauromorphs sampled here (i.e., Gephyrosaurus bridensis: Evans, 1980; Planocephalosaurus robinsonae: Fraser, 1982).

230. Basioccipital, notochordal scar on the occipital surface of the occipital condyle: absent or developed as a small subcircular pit (0); developed as a vertical furrow or a large sub-circular fossa that occupies approximately half of the height of the occipital surface of the condyle (1) (New character). This character is inapplicable in taxa with a concave articular surface of the occipital condyle (Fig. 27).

The presence of a posteriorly opening notochordal scar represented by a subcircular pit on the articular surface of the occipital condyle is a common feature among archosauromorphs. In proterochampsids (e.g., Proterochampsa barrionuevoi: Trotteyn & Haro, 2011; Chanaresuchus bonapartei: PULR 07) and some doswelliids (e.g., Doswellia kaltenbachi: USNM 214823; Jaxtasuchus salomoni: SMNS 91083), the notochordal scar on the occipital condyle is developed as a deep vertical furrow or fossa.

231. Basioccipital, occipital neck: present, distinctly separating the occipital condyle from the basioccipital body (0); absent or extremely short (1) (Ezcurra, Lecuona & Martinelli, 2010: 168) (Fig. 28).

The occipital condyle of the basioccipital lacks a distinct separation from the main body of the bone or is only slightly separated by a transversely and ventrally constricted surface, forming a neck, in several non-archosauriform diapsids (e.g., Simoedosaurus lemoinei: MNHM.F.BL9022, BR10199; Mesosuchus browni: SAM-PK-6536), most proterosuchids (e.g., Proterosuchus goweri: NMQR 880; Proterosuchus alexanderi: NMQR 1484; Proterosuchus fergusi: BP/1/3993) and the doswelliid Archeopelta arborensis (Desojo, Ezcurra & Schultz, 2011). By contrast, the occipital condyle is separated from the base of the basal tubera or the main body of the basioccipital by a distinct neck, namely a constricted non-articular surface that it is as anteroposteriorly long as or longer than the articular surface of the occipital condyle, in Cteniogenys sp. (Evans, 1990), allokotosaurians (Pamelaria dolichotrachela: ISI R316/1; Azendohsaurus madagaskarensis: UA-7-20-99-653; Trilophosaurus buettneri: Spielmann et al., 2008), and most archosauriforms (e.g., see Gower & Sennikov, 1996).

232. Basioccipital, shape of the basal tubera: rounded and anteroposteriorly elongated (0); bladelike and anteroposteriorly shortened (1) (Nesbitt, 2011: 106) (Fig. 28).

See comments in Nesbitt (2011: character 106).

233. Basioccipital, orientation of the basal tubera: lateroventral, basal tubera divergent from each other (0); ventral, basal tubera parallel with each other (1) (New character) (Fig. 27).

The main axis of each basal tuber of the basioccipital is mainly lateroventrally oriented in posterior view, resulting in basal tubera that diverge from each other, in most diapsids. By contrast, in several proterosuchians (e.g., Proterosuchus fergusi: BP/1/3993; Proterosuchus alexanderi: NMQR 1484; Proterosuchus goweri: NMQR 880; Fugusuchus hejiapanensis: (Gower & Sennikov, 1996); an isolated, partial braincase from the Permo-Triassic of Uruguay: Ezcurra et al., 2015) and Proterochampsa barrionuevoi (PVL 2063) the main axes of the basal tubera are vertical and thus these structures extend parallel to each other in posterior view.

234. Parasphenoid-basisphenoid/parabasisphenoid, exposure on the median line of the endocranial cavity floor: present (0); absent (1) (Gower & Sennikov, 1996: 10, 1997; Ezcurra, Lecuona & Martinelli, 2010: 108).

See comments in Gower & Sennikov (1996: character 10).

235. Parasphenoid/parabasisphenoid, orientation: horizontal (0); oblique, main axis posterodorsally-to-anteroventrally oriented (1) (Gower & Sennikov, 1996; Ezcurra, Lecuona & Martinelli, 2010: 28; Nesbitt, 2011: 97; Dilkes & Arcucci, 2012: 32; Nesbitt et al., 2015: 208) (Figs. 27 and 28).

See comments in Gower & Sennikov (1996: character 7) and Nesbitt (2011: character 97).

236. Parasphenoid/parabasisphenoid, posterodorsal portion: incompletely ossified (0); completely ossified (1) (New character).

The lateral surface of the braincase is incompletely ossified, with a gap between the prootic and parasphenoid and posterior to the clinoid process in Youngina capensis (Gardner, Holliday & O’Keefe, 2010) and the choristoderan Simoedosaurus lemoinei (MNHM.F.BL9022, BR10199; Sigogneau-Russell & Russell, 1978). By contrast, the lateral surface of the braincase is completely ossified along the suture between prootic and parabasisphenoid in archosauromorphs.

237. Parasphenoid/parabasisphenoid, intertuberal plate: absent (0); present and posterior edge straight (1); present and posterior edge concave (2) (Gower & Sennikov, 1996, in part; Ezcurra, Lecuona & Martinelli, 2010: 29, in part; Nesbitt, 2011: 96; Dilkes & Arcucci, 2012: 33) (Figs. 10 and 28).

See comments in Gower & Sennikov (1996: character 2) and Nesbitt (2011: character 96).

238. Parasphenoid/parabasisphenoid, semilunar depression on the posterolateral surface of the bone: absent (0); present (1) (Gower & Sennikov, 1996; Ezcurra, Lecuona & Martinelli, 2010: 30; Nesbitt, 2011: 98; Dilkes & Arcucci, 2012: 34; Nesbitt et al., 2015: 209). This character is inapplicable in taxa that the posterodorsal portion of the parasphenoid/parabasisphenoid is not ossified, resulting in an unossified gap between this element and the prootic (Fig. 28).

See comments in Gower & Sennikov (1996: character 11) and Nesbitt (2011: character 98).

239. Parasphenoid/parabasisphenoid, recess (=median pharyngeal recess, =hemispherical sulcus, =hemispherical fontanelle): absent (0); present (1) (Nesbitt & Norell, 2006; Nesbitt, 2011: 100; Dilkes & Arcucci, 2012: 35; Pritchard et al., 2015: 68) (Fig. 27).

See comments in Nesbitt (2011: character 100).

240. Parasphenoid/parabasisphenoid, position of the foramina for entrance of the cerebral branches of the internal carotid artery leading to the pituitary fossa: ventral (0); posterolateral (1); anterolateral (2) (Parrish, 1993; Dilkes, 1998: 45, in part; Ezcurra, Lecuona & Martinelli, 2010: 21, in part; Ezcurra, Scheyer & Butler, 2014: 150, in part; Nesbitt, 2011: 95; Dilkes & Arcucci, 2012: 28; Pritchard et al., 2015: 67) (Fig. 28).

See comments in Gower & Sennikov (1996: character 1) and Nesbitt (2011: character 95).

241. Parasphenoid/parabasisphenoid, position of the foramina for the entrance of the cerebral branches of the internal carotids on the ventral surface of the bone: immediately medial or posteromedial to the base of the basipterygoid process (0); close to the suture between basioccipital and parabasisphenoid (1) (New character). This character is not applicable if foramina for the passage of the internal carotid artery open laterally (Figs. 10 and 28).

The foramina for the entrance of the cerebral branches of the internal carotids, when they are situated on the ventral surface of the braincase, are placed posteromedial to the base of the basipterygoid processes in the vast majority of diapsids scored here. By contrast, in Proterochampsa barrionuevoi (Dilkes & Arcucci, 2012) and Doswellia kaltenbachi (USNM 214823) the entrances of the internal carotids into the endocranial cavity are placed immediately anterior to the suture between the basioccipital and parabasisphenoid on the ventral surface of the braincase. A similar condition to that of the latter two species was originally described for the doswelliid Archeopelta arborensis (Desojo, Ezcurra & Schultz, 2011), but it is here reinterpreted that the entrance of the internal carotids are placed immediately posteromedial to the base of the basipterygoid processes (CPEZ-239a), as in most diapsids.

242. Parasphenoid/parabasisphenoid, shape of the cultriform process in lateral view: continuously tapering anteriorly, without dorsoventral constriction at its base (0); dorsoventrally compressed at its base (1) (Parrish, 1993; Juul, 1994; Gower & Sennikov, 1996; Gower & Sennikov, 1997: 13; Ezcurra, Lecuona & Martinelli, 2010: 110).

See comments in Gower & Sennikov (1996: character 13).

243. Parasphenoid/parabasisphenoid, base of the cultriform process: relatively dorsoventrally short (0); tall, with the dorsal edge extending up between clinoid processes and ventral parts of the crista prootica (1) (Gower & Sennikov, 1996; Gower & Sennikov, 1997: 15; Ezcurra, Lecuona & Martinelli, 2010: 112).

See comments in Gower & Sennikov (1996: character 15).

244. Parasphenoid/parabasisphenoid, dentition on cultriform process: present (0); absent (1) (Gauthier, 1984; Benton, 1985; Pritchard et al., 2015: 65).

245. Basisphenoid/parabasisphenoid, anterior tympanic recess on the lateral side of the braincase: absent (0); present (1) (Makovicky & Sues, 1998; Rauhut, 2003; Nesbitt, 2011: 101) (Fig. 28).

See comments in Nesbitt (2011: character 101).

246. Basisphenoid/parabasisphenoid, parasphenoid crests: absent so that there is no ventral floor for the vidian canal (0); present as a pair of thick crests running along the ventrolateral border of the basisphenoid body and framing the ventromedial floor of the vidian canal (1) (Merck, 1997; Pritchard et al., 2015: 66) (Fig. 10).

See comments in Pritchard et al. (2015: character 66).

247. Basisphenoid/parabasisphenoid, basipterygoid processes: moderately short, finger-like and with short articular facets (0); long, with hemispherical articular facets (1); very short and subcylindrical (2) (Reisz & Dilkes, 2003: 20; Müller, 2004: 96; Reisz, Laurin & Marjanović, 2010: 51; modified from Ezcurra, Scheyer & Butler, 2014: 50).

The basipterygoid process of the basisphenoid/parabasisphenoid in most diapsids is moderately short, finger-like in lateral view, and with distally restricted facets for articulation with the pterygoid. By contrast, the basipterygoid process is proportionally longer and possesses a hemispherical distal articular facet in lepidosauromorphs (e.g., Gephyrosaurus bridensis: Evans, 1980; Planocephalosaurus robinsonae: Fraser, 1982), and it is proportionally very short and subcylidrical in choristoderans (e.g., Cteniogenys sp.: Evans, 1990; Simoedosaurus lemoinei: Sigogneau-Russell & Russell, 1978).

248. Basisphenoid/parabasisphenoid, orientation of basipterygoid processes in the transverse plane: anterolateral or lateral (0); posterolateral (1) (Dilkes, 1998; Ezcurra, Lecuona & Martinelli, 2010: 20; cf. Nesbitt, 2011: 93; Pritchard et al., 2015: 70) (Fig. 28).

This character describes the orientation of the main axis of the basipterygoid process of the parabasisphenoid, not only the orientation of its distal articular end (contrasting with Nesbitt, 2011: 93). In the vast majority of diapsids the basipterygoid processes are anterolaterally or laterally oriented, but in proterosuchids (e.g., Proterosuchus fergusi: BP/1/3993; Proterosuchus goweri: NMQR 880; Proterosuchus alexanderi: NMQR 1484; “Chasmatosaurus” yuani: IVPP V2719), Blomosuchus georgii (PIN 1025/14, 348), Fugusuchus hejiapanensis (Gower & Sennikov, 1996), Garjainia prima (Gower & Sennikov, 1996), Garjainia madiba (Gower et al., 2014), and Parasuchus hislopi (ISI R42) the main axis of these processes are posterolaterally oriented with respect to the transverse plane of the parabasisphenoid.

249. Prootic-supraoccipital, floccular (=auricular) recess: largely restricted to the prootic (0); extends onto internal surface of the supraoccipital (1) (Gower, 2002; Nesbitt, 2011: 133).

See comments in Nesbitt (2011: character 133).

250. Prootic-basisphenoid/parabasisphenoid, position of the external foramina for passage of the abducens nerves (CN VI): within the dorsum sellae (0); track between the dorsum sellae and prootic, grooving the articular facets (1); within the prootic (2) (Evans, 1987; Gower & Sennikov, 1996; Gower, 2002; Nesbitt, 2011: 122; Ezcurra, Scheyer & Butler, 2014: 156, in part; modified from Pritchard et al., 2015: 71). Scored as inapplicable in taxa without distinct abducens nerves foramina.

See comments in Gower & Sennikov (1996: character 3) and Nesbitt (2011: character 122).

251. Prootic-basisphenoid/parabasisphenoid, orientation of the external foramina for passage of the abducens nerves (CN VI): open anteriorly (0); open dorsally (1) (Gower & Sennikov, 1996: 4; Nesbitt, 2011: 123). Scored as inapplicable in taxa without distinct abducens nerves foramina.

See comments in Gower & Sennikov (1996: character 4) and Nesbitt (2011: character 123).

252. Prootic, extensive contact with parietal: absent (0); present (1) (DeBraga & Rieppel, 1997; Müller, 2004: 160; Ezcurra, Scheyer & Butler, 2014: 155).

253. Prootic, contact with its counterpart on the median line of the floor of the endocranial cavity: absent (0); present (1) (Gower & Sennikov, 1996: 9, Gower & Sennikov, 1997; Ezcurra, Lecuona & Martinelli, 2010: 107).

See comments in Gower & Sennikov (1996: character 9).

254. Prootic, lateral surface: continuous and slightly convex (0); crista prootica present (1) (Dilkes, 1998: 47; Ezcurra, Lecuona & Martinelli, 2010: 106, in part; Ezcurra, Scheyer & Butler, 2014: 153; Pritchard et al., 2015: 73) (Fig. 28).

The ventrolateral margin of the prootic is rather flat and confluent with the lateral surface of the parabasisphenoid in Youngina capensis (Gardner, Holliday & O’Keefe, 2010) and Simoedosaurus lemoinei (Sigogneau-Russell & Russell, 1978). In all the archosauromorphs sampled here, the ventrolateral margin of the prootic is laterally projected and overhangs the lateral surface of the parabasisphenoid, forming an anteroventrally-to-posterodorsally oriented crista prootica.

255. Prootic, anterior inferior process: absent or developed as a small, peg-like projection (0); well developed (1) (Dilkes, 1998: 48; Ezcurra, Scheyer & Butler, 2014: 154; modified from Pritchard et al., 2015: 74) (Fig. 28).

The external opening for the passage of CN V is formed by subequally developed inferior and superior anterior processes in most archosauromorphs (Evans, 1986; Gower & Sennikov, 1996). By contrast, the anterior inferior process is absent in Simoedosaurus lemoinei (Sigogneau-Russell & Russell, 1978) and Tanystropheus longobardicus (Wild, 1973), or reduced to a small, peg-like projection in Youngina capensis (Gardner, Holliday & O’Keefe, 2010).

256. Prootic, ridge on the lateral surface of the inferior anterior process ventral to the trigeminal foramen: present (0); absent (1) (Gower & Sennikov, 1996; Ezcurra, Lecuona & Martinelli, 2010: 24; Nesbitt, 2011: 94; Dilkes & Arcucci, 2012: 31). This character is scored as inapplicable in taxa that lack an inferior anterior process (Fig. 28).

See comments in Gower & Sennikov (1996: character 6) and Nesbitt (2011: character 94).

257. Prootic, medial surface of vestibule: incompletely ossified (0); almost completely ossified (1) (Gower, 2002; Nesbitt, 2011: 117).

See comments in Nesbitt (2011: character 117).

258. Laterosphenoid, ossification: absent (0); present (1) (Dilkes, 1998: 50; Ezcurra, Lecuona & Martinelli, 2010: 26; Nesbitt, 2011: 92; Ezcurra, Scheyer & Butler, 2014: 149; Pritchard et al., 2015: 72) (Fig. 28).

See comments in Nesbitt (2011: character 92).

259. Laterosphenoid, anterodorsal channel: absent (0); present (1) (Gower & Sennikov, 1996; Gower & Sennikov, 1997; Ezcurra, Lecuona & Martinelli, 2010: 109). Character inapplicable in taxa lacking an ossified laterosphenoid.

See comments in Gower & Sennikov (1996: character 12).

260. Lower jaw, symphysis: formed largely by dentary (0), formed only by splenial (1) (Dilkes, 1998; Pritchard et al., 2015: 78).

In the vast majority of diapsids the contact between the hemimandibles is largely restricted to the dentaries. In rhynchosaurids (e.g., Rhynchosaurus articeps: NHMUK PV R1236, R1237; Bentonyx sidensis: BRSUG 21200) the dentaries diverge from each other at the anterior ends of the hemimandibles and the splenials form the mandibular symphysis by a dorsoventrally extensive median contact.

261. Lower jaw, distinct dorsal process behind the alveolar margin: absent, with a slightly convex dorsal margin behind the alveolar portion (0); present, formed by a dorsally well-developed surangular (1); present, formed by a dorsally well-developed posterodorsal ramus of the dentary and sometimes a dorsally well-developed coronoid bone (2) (Rieppel, Mazin & Tchernov, 1999; Müller, 2004: 36; Nesbitt, 2011: 158, in part; modified from Ezcurra, Scheyer & Butler, 2014: 158; Pritchard et al., 2015: 79) (Fig. 29).

Figure 29 Lower jaws of Triassic archosauromorphs in (A, E) medial and (B–D, F) lateral views.

(A) Left hemimandible of Pamelaria dolichotrachela (ISI R316/1); (B) reconstruction of Prolacerta broomi; (C) reconstruction of Proterosuchus fergusi; (D) reconstruction of Garjainia prima; (E) posterior half of the left hemimandible of Garjainia prima (PIN 951/33); and (F) right dentary of “Chasmatosaurus” yuani (IVPP V36315). Numbers indicate character-states scored in the data matrix and the arrows indicate anterior direction. Scale bars equal 1 cm in (A, E, F) and B–D not to scale.

See comments in Nesbitt (2011: character 158). Taxa with a well-developed dorsal process posterior to the alveolar margin of the lower jaw, formed by the surangular include rhynchosaurids (i.e., Rhynchosaurus articeps: NHMUK PV R1236; Bentonyx sidensis: BRSUG 21200) and some phytosaurs (e.g., Nicrosaurus kapffi: NHMUK PV R38036; Smilosuchus gregorii: UCMP 27200). By contrast, the dentary and sometimes a coronoid bone contribute to this dorsal expansion of the lower jaw in lepidosauromorphs (Gephyrosaurus bridensis: Evans, 1980; Planocephalosaurus robinsonae: 1982), some allokotosaurians (e.g., Pamelaria dolichotrachela: ISI R316; Trilophosaurus buettneri: Spielmann et al., 2008), and the archosaurs Aetosauroides scagliai (PVL 2073) and Heterodontosaurus tucki (SAM-PK-K337, K1332).

262. Lower jaw, external mandibular fenestra: absent (0); present (1) (Dilkes, 1998: 76; Ezcurra, Lecuona & Martinelli, 2010: 42; Nesbitt, 2011: 138; Ezcurra, Scheyer & Butler, 2014: 166; Pritchard et al., 2015: 84) (Figs. 17 and 29).

See comments in Nesbitt (2011: character 138).

263. Lower jaw, anteroposterior length of the external mandibular fenestra versus anteroposterior length of the dentary anterior to the fenestra: 0.07–0.36 (0); 0.44–0.53 (1); 0.71–0.88 (2) (Butler, 2005; modified from Nesbitt, 2011: 162; cf. Dilkes & Arcucci, 2012: 47), ORDERED. This character is inapplicable in taxa that lack an external mandibular fenestra (Figs. 18 and 29).

See comments in Nesbitt (2011: character 162).

264. Lower jaw, Meckelian fossa orientation: dorsomedially (0); mostly dorsally due to greatly expanded prearticular resulting in a ventral border of the fossa situated dorsal to the half-height of the lower jaw at that level (1) (DeBraga & Rieppel, 1997; Müller, 2004: 165; Ezcurra, Scheyer & Butler, 2014: 159) (Fig. 29).

The prearticular forms the medial wall of the Meckelian fossa and, in most diapsids sampled here, it is strongly dorsoventrally constricted close to its mid-length. As a result, the Meckelian fossa is widely visible in medial view and opens dorsomedially. By contrast, in the allokotosaurian Trilophosaurus buettneri (Spielmann et al., 2008) and in the phytosaurs Nicrosaurus kapffi (NHMUK PV R42744) and Smilosuchus gregorii (UCMP 27200) the prearticular is slightly dorsoventrally compressed, remaining as a tall bone along its extension and, as a consequence, the Meckelian fossa opens mainly dorsally.

265. Dentary-splenial, mandibular symphysis length: positioned distally (0); present along one-third of the lower jaw (1) (Sereno, 1991; Nesbitt, 2011: 160).

See comments in Nesbitt (2011: character 160).

266. Dentary, minimum height of the bone versus length of the alveolar margin (including edentulous anterior end if present): 0.05–0.14 (0); 0.16–0.19 (1); 0.22–0.29 (2); 0.34–0.36 (3) (New character), ORDERED (Figs. 17 and 18).

This character accounts for the relative dorsoventral depth of the dentary at the point of its lowest dorsoventral depth in lateral view in comparison with the length of the bone from its anterior tip to the posterior margin of the last alveolous. The dentary is a very low bone in several archosauriform lineages, such as proterochampsids, phytosaurs, and some basal suchians (ratio < 0.15). Conversely, the dentary is deep in Riojasuchus tenuisceps (Bonaparte, 1971) and Vancleavea campi (Nesbitt et al., 2009) (ratio > 0.30). Most diapsids scored here exhibit intermediate conditions.

267. Dentary, shape of the tooth bearing portion: mostly straight (0); distinctly dorsally curved for all or most of its anteroposterior length (1); ventrally curved or deflected (2) (modified from Nesbitt, 2011: 154, in part, and Nesbitt et al., 2015: 241, in part) (Figs. 17 and 29).

See comments in Nesbitt (2011: character 154).

268. Dentary, large foramina aligned in two distinct rows starting on the anteroventral corner of the bone: absent (0); present (1) (New character) (Fig. 29).

The lateral surface of the anterior end of the dentary commonly possesses multiple foramina in most diapsids. In particular, large foramina are aligned in two distinct rows that extend longitudinally on the anteroventral corner of the bone in Petrolacosaurus kansensis (Reisz, 1981), Youngina capensis (SAM-PK-K7578), Cteniogenys sp. (Evans, 1990), Pamelaria dolichotrachela (ISI R316/1), Jesairosaurus lehmani (ZAR 06), Prolacerta broomi (BP/1/2675), proterosuchids (e.g., Proterosuchus fergusi: BP/1/3993; SAM-PK-K11208; Proterosuchus alexanderi: NMQR 1484; “Chasmatosaurus” yuani: IVPP V4067, V36315), Sarmatosuchus otschevi (PIN 2865/68), Rhadinosuchus gracilis (BSPG AS XXV 50), and a few archosaurs (e.g., Nundasuchus songeaensis: Nesbitt et al., 2014).

269. Dentary, longitudinal groove approximately centred dorsoventrally on the lateral surface: absent (0); present (1) (New character) (Fig. 18).

In the vast majority of diapsids the lateral surface of the dentary is smooth, but there is a longitudinal, well defined groove in Pamelaria dolichotrachela (ISI R316/1), some proterochampsids (e.g., Chanaresuchus bonapartei: PULR 07; Rhadinosuchus gracilis: BSPG AS XXV 50), phytosaurs (e.g., Parasuchus hislopi: ISI R42; Smilosuchus georgii: UCMP 27200), Ornithosuchus longidens (NHMUK PV R3143), Nundasuchus songeaensis (Nesbitt et al., 2014), most basal suchians (e.g., Gracilisuchus stipanicicorum: MCZ 4117; Turfanosuchus dabanensis: IVPP V3237; Aetosauroides scagliai: PVL 2052; Batrachotomus kupferzellensis: SMNS 80260), and some avemetatarsalians (e.g., Dimorphodon macronyx: NHMUK PV R41212-13; Herrerasaurus ischigualastensis: PVSJ 407).

270. Dentary, position of the Meckelian groove on the anterior half of the bone: dorsoventral centre of the dentary (0); restricted to the ventral border (1) (Nesbitt, 2011: 152) (Fig. 27 of Nesbitt, 2011).

See comments in Nesbitt (2011: character 152).

271. Dentary, anterior portion: unexpanded, dorsal margins of the anterior and posterior portions of the bone in the same plane (0); dorsally expanded, whole dorsoventral height of the anterior portion is greater than that of the posterior portion (1) (modified from Nesbitt, 2011: 154) (Figs. 18, 19 and 29).

See comments in Nesbitt (2011: character 154).

272. Dentary, posterodorsal process, in which its dorsal margin is confluent with the dorsal margin of the lower jaw: absent (0); present (1) (New character) (Figs. 17 and 29).

In the rhynchocephalians Gephyrosaurus bridensis and Planocephalosaurus robinsonae, the dentary forms approximately half of the dorsal margin of the lower jaw posterior of the alveolar margin (Evans, 1980; Fraser, 1982). By contrast, the extent of the dentary on the dorsal margin of the lower jaw is only very limited in lateral view in other diapsids.

273. Dentary, posterocentral process, in which its margins are not confluent with the dorsal or ventral margin of the lower jaw: absent (0); present (1) (New character; cf. Nesbitt et al., 2015: 246) (Figs. 17 and 29).

The dentary possesses a posterior process that it is centred approximately at mid-height of the posterior end of the bone and is not confluent with the dorsal and ventral margins of the lower jaw (i.e., posterocentral process) in lepidosauromorphs (e.g., Planocephalosaurus robinsonae: Fraser, 1982; Gephyrosaurus bridensis: Evans, 1980), choristoderans (e.g., Cteniogenys sp.: Evans, 1990; Simoedosaurus lemoinei: Sigogneau-Russell & Russell, 1978), Tasmaniosaurus triassicus (UTGD 54655), proterosuchids (e.g., Proterosuchus fergusi: SAM-PK-11208, RC 59; Proterosuchus alexanderi: NMQR 1484; “Chasmatosaurus” yuani: IVPP V4067), erythrosuchids (e.g., Garjainia prima: PIN 2394/5; Erythrosuchus africanus: BP/1/5207; Shansisuchus shansisuchus: Young, 1964), Euparkeria capensis (SAM-PK-5867), most suchians (e.g., Turfanosuchus dabanensis: IVPP V3237; Batrachotomus kupferzellensis: Gower, 1999; Prestosuchus chiniquensis: UFRGS-PV-0156-T), and some dinosauriforms (e.g., Silesaurus opolensis: ZPAL AbIII/1930/92/25; Herrerasaurus ischigualastensis: PVSJ 407). The posterocentral process forms the anterodorsal border of the external mandibular fenestra in taxa that possess this opening.

274. Dentary, distal end of the posterocentral process: tapering (0); rounded (1) (New character). This character is inapplicable in taxa that lack a posterocentral process in the dentary (Figs. 17 and 29).

Among the diapsids sampled here that possess a posterocentral process, only lepidosauromorphs (e.g., Planocephalosaurus robinsonae: Fraser, 1982; Gephyrosaurus bridensis: Evans, 1980), the choristoderan Cteniogenys sp. (Evans, 1990), and proterosuchids (e.g., Proterosuchus fergusi: SAM-PK-11208, RC 59; Proterosuchus alexanderi: NMQR 1484; “Chasmatosaurus” yuani: IVPP V4067) exhibit a rounded distal end of this process. In other species, the posterocentral process tapers continuously towards its distal end.

275. Dentary, posteroventral process, in which its ventral margin is confluent with the ventral margin of the lower jaw: absent (0); present and excluded from the anteroventral border of the external mandibular fenestra (1); present and contributing to the anteroventral border of the external mandibular fenestra (2) (reworded and modified from Nesbitt et al., 2009; Nesbitt, 2011: 164; Nesbitt et al., 2015: 210, in part) (Figs. 17 and 29).

See comments in Nesbitt (2011: character 164).

276. Dentary, posteroventral process length: extended posteriorly to the level of the posterodorsal and/or posterocentral processes (0); extended posteriorly beyond the level of the posterodorsal and/or posterocentral processes (1) (reworded from Nesbitt et al., 2015: 246). This character is inapplicable in taxa that lack a posteroventral process in the dentary (Fig. 29).

See comments in Nesbitt et al. (2015: character 246).

277. Posteriormost dentary teeth: on the anterior half of lower jaw (0); on the posterior half of lower jaw (1) (Langer & Schultz, 2000) (Fig. 17).

The alveolar margin of the lower jaw extends along more than half of the hemimandibular length in most non-archosauromorph diapsids sampled here (e.g., Petrolacosaurus kansensis: Reisz, 1981; Gephyrosaurus bridensis: Evans, 1980; Simoedosaurus lemoinei: MNHN.F.R4014) and Protorosaurus speneri Gottmann-Quesada & Sander, 2009, Tanystropheus longobardicus (Wild, 1973; Nosotti, 2007), the allokotosaurians Azendohsaurus madagaskarensis (Flynn et al., 2010) and Trilophosaurus buettneri (Spielmann et al., 2008), phytosaurs (e.g., Parasuchus hislopi: ISI R42; Nicrosaurus kapffi: NHMUK PV R38036; Smilosuchus gregorii: UCMP 27200), Dimorphodon macronyx (NHMUK PV R41212-13), and Heterodontosaurus tucki (SAM-PK-K337, K1332). By contrast, in most non-archosaurian archosauriforms, the alveolar margin of the dentary is restricted to the anterior half of the lower jaw.

278. Dentary, alveolar margin: present along entire length of the dentary (0); absent in the anterior portion (1) (Parrish, 1994; modified from Nesbitt, 2011: 166; Pritchard et al., 2015: 88, in part) (Fig. 29).

See comments in Nesbitt (2011: character 166).

279. Dentary, number of tooth rows: one (0); two (1); more than two (2) (Dilkes, 1998: 64; Ezcurra, Scheyer & Butler, 2014: 160), ORDERED.

The dentary possesses a single row of marginal teeth in the vast majority of diapsids, but in rhynchosaurs there are two or more rows of teeth in each hemimandible (Chatterjee, 1980; Benton, 1984b; Dilkes, 1998; Ezcurra, Montefeltro & Butler, 2016).

280. Dentary, occlusion with upper teeth: single-sided overlap (0); flat occlusion (1); blade and groove (2) (Dilkes, 1998: 65; Ezcurra, Scheyer & Butler, 2014: 161) (Fig. 14).

The ancestral diapsid condition is the presence of a single-sided occlusion between the upper and lower marginal teeth. In non-rhynchosaurid rhynchosaurs (e.g., Mesosuchus browni: Dilkes, 1998; Howesia browni: Dilkes, 1995; Eohyosaurus wolvaardti: Butler et al., 2015) the cranial teeth possess a flat, broad occlusion with the lower teeth, whereas in rhynchosaurids the occlusion is specialized in a blade and groove system present in the dentary and maxilla, respectively (Chatterjee, 1980; Benton, 1984b; Dilkes, 1998).

281. Surangular-angular, suture: even with lateral surface of hemimandible (0); elevated and separates dorsal concave area on surangular from concave area on angular (1) (Dilkes & Arcucci, 2012: 45).

The anterior half of the suture between the surangular and angular runs along a thick and low tuberosity placed immediately posterior of the external mandibular fenestra on the lateral surface of the hemimandible in the proterochampsids Tropidosuchus romeri (PVL 4606), Gualosuchus reigi (PVL 4576) and Chanaresuchus bonapartei (PULR 07). By contrast, the suture between the surangular and angular is even with the rest of the lateral surface of the posterior half of the hemimandible in other diapsids.

282. Surangular-angular, suture along the anterior half of the bones in lateral view: anteroposteriorly convex ventrally (0); anteroposteriorly concave ventrally (1) (New character) (Fig. 29).

283. Surangular-articular, retroarticular process: absent (0); anteroposteriorly short, being poorly developed posteriorly to the glenoid fossa (1); anteroposteriorly long, extending considerably posterior to the glenoid fossa (2) (Laurin, 1991: B6, E10, J5; Reisz, Laurin & Marjanović, 2010: 63; reworded from Ezcurra, Scheyer & Butler, 2014: 62; Trotteyn & Ezcurra, 2014: 109, in part; Pritchard et al., 2015: 86, in part), ORDERED (Figs. 17 and 29).

The retroarticular process of the hemimandible is absent in the early diapsid Petrolacosaurus kansensis (Reisz, 1981) and the proterochampsids Proterochampsa barrionuevoi (MACN-Pv 18165; PVL 2058, 2061), Proterochampsa nodosa (MCP 1694) and Pseudochampsa ischigualastensis (Trotteyn & Ezcurra, 2014). In other diapsids, the retroarticular process possesses different degrees of posterior development that are here qualitatively discretized.

284. Surangular-articular, retroarticular process: not upturned (0); upturned (1) (Dilkes, 1998: 75; Müller, 2004: 101; Ezcurra, Scheyer & Butler, 2014: 167). This character is scored as inapplicable in taxa that lack a retroarticular process (Figs. 17 and 29).

The retroarticular process of the hemimandible is straight in lateral view in non-archosauromorph neodiapsids, Protorosaurus speneri (Gottmann-Quesada & Sander, 2009), tanystropheids (e.g., Amotosaurus rotfeldensis: SMNS unnumbered; Tanystropheus longobardicus: PIMUZ T2189) and some archosauriforms (e.g., Doswellia kaltenbachi: USNM 214823; Nicrosaurus kapffi: NHMUK PV R38036; Riojasuchus tenuisceps: PVL 3827; Heterodontosaurus tucki: SAM-PK-K337). By contrast, the retroarticular process is dorsally curved in lateral view in allokotosaurians (e.g., Pamelaria dolichotrachela: ISI R316/1; Trilophosaurus buettneri: Spielmann et al., 2008), rhynchosaurs (e.g., Mesosuchus browni: SAM-PK-6536; Eohyosaurus wolvaardti: SAM-PK-K10159), Prolacerta broomi (SAM-PK-K10018, K10797), and most archosauriforms (e.g., Proterosuchus alexanderi: NMQR 1484; Chanaresuchus bonapartei: MCZ 4037; Parasuchus hislopi: ISI R42; Herrerasaurus ischigualastensis: PVSJ 407).

285. Surangular, anterior extension: beyond coronoid eminence (0); posterior to reaching the anterior border of the coronoid eminence (1) (DeBraga & Rieppel, 1997; Müller, 2004: 143; Ezcurra, Scheyer & Butler, 2014: 162) (Fig. 29).

The surangular is poorly anteriorly extended in lateral view, without reaching the level of the anterior end of the dorsally raised portion of the lower jaw immediately posterior of the alveolar margin (i.e., coronoid eminence) in lepidosauromorphs (e.g., Gephyrosaurus bridensis: Evans, 1980; Planocephalosaurus robinsonae: Fraser, 1982), Trilophosaurus buettneri (Spielmann et al., 2008) and Nicrosaurus kapffi (NHMUK PV R38036). In other diapsids scored here, the suangular is more anteriorly extended on the lateral surface of the hemimandible.

286. Surangular, lateral shelf: absent (0); present, low ridge near dorsal margin (1); present, presence of laterally or ventrolaterally projecting shelf with straight or gently convex lateral edge (2); present, presence of laterally projecting shelf with strongly convex lateral edge (3) (DeBraga & Rieppel, 1997; Müller, 2004: 166; Dilkes & Arcucci, 2012: 43, in part; Ezcurra, Scheyer & Butler, 2014: 163) (Figs. 18 and 29).

See comments in Dilkes & Arcucci (2012: character 43).

287. Surangular, dorsal margin in lateral view: straight or gently convex (0); strongly convex (1) (Dilkes & Arcucci, 2012: 44) (Fig. 29).

See comments in Dilkes & Arcucci (2012: character 44).

288. Surangular, anterior surangular foramen on the lateral surface of the bone, near surangular-dentary contact: absent (0); present (1) (Modesto & Sues, 2004: 145; Ezcurra, Lecuona & Martinelli, 2010: 43; Dilkes & Arcucci, 2012: 48; Ezcurra, Scheyer & Butler, 2014: 164; Pritchard et al., 2015: 80) (Fig. 29).

See comments in Modesto & Sues (2004: 348).

289. Surangular, posterior surangular foramen on the lateral surface of the bone, positioned directly anterolateral to the glenoid fossa: absent (0); present (1) (Modesto & Sues, 2004: 146; Ezcurra, Lecuona & Martinelli, 2010: 44; Nesbitt, 2011: 163, in part; Dilkes & Arcucci, 2012: 49; Ezcurra, Scheyer & Butler, 2014: 165; Pritchard et al., 2015: 81) (Fig. 29).

This character-state is represented by character 163 of Nesbitt (2011), and all non-archosaurian archosauromorphs were scored as having a posterior surangular foramen in his analysis (see also comments in Modesto & Sues (2004: 348). However, a broader variability regarding the absence and presence of this foramen is found among basal archosauromorphs in the current phylogenetic analysis. The presence of a posterior surangular foramen is recovered here as plesiomorphic for Archosauriformes, being retained by proterosuchids, some erythrosuchids and Euparkeria capensis. By contrast, this foramen is absent in proterochampsids (e.g., Proterochampsa barrionuevoi: Dilkes & Arcucci, 2012; Gualosuchus reigi: PULR 05, PVL 4576; the condition is polymorphic in Chanaresuchus bonapartei, being present in MCZ 4037 and absent in PULR 07 and PVL 4586), doswelliids (Doswellia kaltenbachi: USNM 214823), and several archosaurs (e.g., Parasuchus hislopi: ISI R42; Nundasuchus songeaensis: Nesbitt et al., 2014; Turfanosuchus dabanensis: IVPP V3237; Gracilisuchus stipanicicorum: MCZ 4117).

290. Angular, dorsoventral exposure on the lateral surface of the lower jaw: wide (0); narrow (1) (Laurin, 1991: J4; Müller, 2004: 167; Reisz, Laurin & Marjanović, 2010: 62; Ezcurra, Scheyer & Butler, 2014: 61; Pritchard et al., 2015: 82) (Fig. 29).

This character distinguishes between a widely exposed angular on the posterior half of the lower jaw in lateral view, with a subequal contribution to the dorsoventral height of the hemimandible posteriorly to the external mandibular fenestra, and a dorsoventrally narrow angular that is restricted to the lateroventral surface of the lower jaw. A narrow angular in lateral view is present in various diapsid species scored here, including Gephyrosaurus bridensis (Evans, 1980), Simoedosaurus lemoinei (MNHN.F.R4014), rhynchosaurs (e.g., Mesosuchus browni: SAM-PK-6536; Rhynchosaurus articeps: SHYMS 1), Prolacerta broomi (SAM-PK-K10797), Euparkeria capensis (SAM-PK-5867), and some phytosaurs (e.g., Parasuchus hislopi: ISI R42).

291. Angular, ventrolateral surface: continuous with lateral surface of angular (0); laterally projecting ridge present that separates lateral and ventral sides of the angular (1) (Dilkes & Arcucci, 2012: 46).

See comments in Dilkes & Arcucci (2012: character 46).

292. Angular, posteroventral surface: ridged or keeled (0); transversely convex (1) (Reisz & Dilkes, 2003: 38; Reisz, Laurin & Marjanović, 2010: 61; Ezcurra, Scheyer & Butler, 2014: 60; Trotteyn & Ezcurra, 2014: 110).

The posteroventral surface of the angular is continuously transversely convex in the vast majority of diapsids, but in Proterochampsa barrionuevoi and Proterochampsa nodosa there is a longitudinal ridge on this surface of the angular (Dilkes & Arcucci, 2012).

293. Articular, fused to the prearticular: absent (0); present (1) (Benton, 1985; Pritchard et al., 2015: 87).

The articular is fused to the prearticular in the lepidosauromorphs Gephyrosaurus bridensis (Evans, 1980) and Planocephalosaurus robinsonae (Fraser, 1982), contrasting with the condition in other diapsids.

294. Articular, foramen on the medial side: absent (0); present (1) (Nesbitt, 2011: 159) (Fig. 26 of Nesbitt, 2011).

See comments in Nesbitt (2011: character 159).

295. Articular, ventromedially directed process: absent (0); present (1) (Nesbitt, 2011: 157) (Fig. 29).

See comments in Nesbitt (2011: character 157).

296. Stapes, shape: robust, with thick shaft (0); slender, rod-like shaft (1) (Laurin, 1991: E8; Reisz, Laurin & Marjanović, 2010: 65; Ezcurra, Scheyer & Butler, 2014: 64; Pritchard et al., 2015: cf. 76).

The stapes is a thin, rod-like bone in archosauromorphs. By contrast, the stapes is considerably more robust and resembles in size and thickness the paroccipital process of the opisthotic in the basal diapsids Petrolacosaurus kansensis (Reisz, 1981) and Youngina capensis (Gardner, Holliday & O’Keefe, 2010).

297. Stapes, stapedial foramen piercing the columellar process: present (0); absent (1) (Laurin, 1991: E9; Reisz, Laurin & Marjanović, 2010: 66; Ezcurra, Scheyer & Butler, 2014: 65; Pritchard et al., 2015: 77).

The proximal portion of the stapes is pierced by an oval stapedial foramen in Petrolacosaurus kansensis (Reisz, 1981) and Youngina capensis (Gardner, Holliday & O’Keefe, 2010), but this opening is absent in archosauromorphs.

298. Teeth, posterior extent of mandibular and maxillary tooth rows: subequal (0); maxillary teeth extending further posteriorly (1) (Bennett, 1996; Ezcurra, Lecuona & Martinelli, 2010: 36).

The dentition of the maxilla extends farther posteriorly than that of the dentary in most diapsids, but in Petrolacosaurus kansensis (Reisz, 1981), rhynchosaurs (e.g., Mesosuchus browni: SAM-PK-6536; Rhynchosaurus articeps: NHMUK PV R1236), Nicrosaurus kapffi (NHMUK PV R38036), and Heterodontosaurus tucki (SAM-PK-K337, K1332) the upper and lower alveolar margins extend posteriorly to about the same extent.

299. Teeth, tooth implantation: subthecodont (=protothecodont) (0); ankylothecodont (teeth fused to the bone at the base of the crown by bony ridges and the root can be discerned; there is continuous tooth replacement) (1); pleurodont (2); acrodont (teeth fused to the bone in adults so that no root can be discerned) (3); thecodont (4) (Dilkes, 1998: 55; Laurin, 1991: G4; Müller, 2004: 38; modified from Reisz, Laurin & Marjanović, 2010: 1; Ezcurra, Lecuona & Martinelli, 2010: 102, in part; Nesbitt, 2011: 174, in part; Ezcurra, Scheyer & Butler, 2014: 1; Pritchard et al., 2015: 94, 95, 97) (Figs. 12, 14 and 22).

See comments in Modesto & Sues (2004: 347) and Nesbitt (2011: character 174).

300. Teeth, maxillary and/or dentary tooth crowns: generally homodont (0); markedly heterodont (gross change in morphology) (1) (Parrish, 1993; Nesbitt, 2011: 167) (Fig. 14).

See comments in Nesbitt (2011: character 167).

301. Teeth, enlarged caniniform region in maxilla: present (0); absent (1) (Benton, 1985; Pritchard et al., 2015: 89) (Fig. 18).

The presence of a caniniform region composed of some consecutive maxillary tooth crowns that are apicobasally taller than others is a common condition among basal amniotes (e.g., Petrolacosaurus kansensis: Reisz, 1981). This feature was here recognized only in the bizarre archosauriform Vancleavea campi (USNM 508519, cast of GR 138) among the neodiapsids sampled here.

302. Teeth, maxillary tooth crowns in labial view: all the tooth crowns possess a rather similar distal edge morphology along the entire alveolar margin (0); the distal edge of the posterior tooth crowns possess a distinct different morphology from those of the anterior tooth crowns, with the posterior edge usually convex (1) (modified from Sues et al., 2003; modified from Nesbitt, 2011: 15) (Fig. 18).

See comments in Nesbitt (2011: character 15).

303. Teeth, distal edge of the maxillary tooth crowns in labial view: concave in all tooth crowns (0); straight or gently sigmoid (1); convex in at least some anterior tooth crowns (2) (Reisz & Dilkes, 2003: 1; Reisz, Laurin & Marjanović, 2010: 2; Ezcurra, Lecuona & Martinelli, 2010: 34, in part; Nesbitt, 2011: 173, in part; modified from Ezcurra, Scheyer & Butler, 2014: 2; Pritchard et al., 2015: 91, in part). This character is not applicable to taxa with the posterior edge of the posterior tooth crowns different from those of the anterior tooth crowns or that possess multiple tooth rows in the maxilla (Fig. 14).

See comments in Nesbitt (2011: character 173).

304. Teeth, serrations on the maxillary/dentary crowns: absent (0); distinctly present on the distal margin and usually apically restricted, low or absent on the mesial margin (1); present and distinct on both margins (2) (Dilkes, 1998; Reisz & Dilkes, 2003: 32; Reisz, Laurin & Marjanović, 2010: 3; Ezcurra, Lecuona & Martinelli, 2010: 33,modified; Nesbitt, 2011: 168, in part, modified; Ezcurra, Scheyer & Butler, 2014: 3; Pritchard et al., 2015: 90) (Fig. 14).

See comments in Nesbitt (2011: character 168).

305. Teeth, labiolingual compression of the marginal dentition: only distally or nowhere (0); present (1) (Dilkes, 1998; Reisz & Dilkes, 2003:1, 34; Reisz, Laurin & Marjanović, 2010: 4; Ezcurra, Lecuona & Martinelli, 2010: 35; Ezcurra, Scheyer & Butler, 2014: 4; Pritchard et al., 2015: 98) (Fig. 14).

The tooth crowns of the marginal dentition are subcircular in cross-section in most non-archosauriform diapsids (e.g., Simoedosaurus lemoinei: MNHN.F.BR1935; Macrocnemus bassanii: PIMUZ T4822; Prolacertoides jimusarensis: IVPP V3233; Mesosuchus browni: SAM-PK-6536), with the exception of some specimens of Youngina capensis (GHG RS 160, GHG K106) and Planocephalosaurus robinsonae (Fraser, 1982), the allokotosaurians Pamelaria robinsonae (ISI R316/1) and Azendohsaurus madagaskarensis (UA 8-29-97-160, 10603, 10604), and Prolacerta broomi (BP/1/2675). Conversely, the vast majority of Permo-Triassic archosauriforms possess labiolingually compressed tooth crowns with an oval cross-section (Dilkes, 1998). Among the archosauriforms sample here, only Proterochampsa nodosa (MCP 1694), Doswellia kaltenbachi (USNM 214823), Smilosuchus gregorii (UCMP 27200), and Silesaurus opolensis (ZPAL AbIII/361/26, 27, 1216; ZPAL AbIII/1930/92/25) lack labiolingually uncompressed crowns throughout the entire marginal dentition. This condition is restricted to some teeth in Nicrosaurus kapffi (Hungerbühler, 2000) and Heterodontosaurus tucki (SAM-PK-K337, K1332) and, as a result, the character is scored as polymorphic.

Figure 30 Anterior cervical vertebrae of Triassic archosauromorphs in left lateral view.

(A) Tanystropheus longobardicus (PIMUZ T2189); (B) Sarmatosuchus otschevi (PIN 2865/68-22); (C) Parasuchus hislopi (ISI R42); and (D) Silesaurus opolensis (ZPAL AbIII unknown number). Numbers indicate character-states scored in the data matrix and the arrows indicate anterior direction. Scale bars equal 2 cm in (A), 5 mm in (B, C), and 2 mm in (D).

306. Teeth, multiple maxillary or dentary tooth crowns with longitudinal labial or lingual striations or grooves: absent (0); present (1) (New character) (Fig. 14).

The surface of the enamel of the tooth crowns of most diapsids is unornamented, but longitudinal (apicobasally oriented) striations or grooves are sporadically observed in the taxonomic sample of the present analysis, being present in Petrolacosaurus kansensis (Reisz, 1981), Cteniogenys sp. (Evans, 1990), Amotosaurus rotfeldensis (SMNS unnumbered), Tanystropheus longobardicus (SMNS 54147), Jaxtasuchus salomoni (SMNS 91083), Smilosuchus gregorii (UCMP 27200), Silesaurus opolensis (Dzik, 2003), and Heterodontosaurus tucki (SAM-PK-K337, K1332).

307. Teeth, multiple maxillary and dentary tooth crowns with extensive wear facets: absent (0); present (1) (Weishampel & Witmer, 1990; Nesbitt, 2011: 169) (Fig. 14).

See comments in Nesbitt (2011: character 169).

308. Teeth, multiple maxillary and dentary tooth crowns distinctly mesiodistally expanded above the root: absent (0); present (1) (Sereno, 1986; Nesbitt, 2011: 171) (Fig. 14).

See comments in Nesbitt (2011: character 171).

309. Hyoid apparatus, length and orientation of the ceratobranchial: short, directed to quadrate region (0); long, directed posteriorly and extending posteriorly beyond the quadrate condyles (1) (Reisz & Dilkes, 2003: 40; Reisz, Laurin & Marjanović, 2010: 67; Ezcurra, Scheyer & Butler, 2014: 66).

The ceratobranchial bones of the hyoid apparatus are short and do not extend posteriorly to the level of the craniomandibular joint in Petrolacosaurus kansensis (Reisz, 1981) and Acerosodontosaurus piveteaui (MNHN 1908-32-57). By contrast, in all the archosauromorphs sampled here the ceratobranchials are proportionally longer and extend posteriorly beyond the level of the distal condyles of the quadrate.

310. Cervical, dorsal, sacral and caudal vertebrae, notochordal canal piercing the centrum: present throughout ontogeny (0); absent in adults (1) (Laurin, 1991: F3; Reisz, Laurin & Marjanović, 2010: 68; Ezcurra, Scheyer & Butler, 2014: 67) (Fig. 31).

Figure 31 Presacral vertebrae of Permo-Triassic and Recent saurians in (A–E) posterior, (F–H) lateral, and (I, J) dorsal views.

(A) Salvator merianae (MACN-He 47992); (B) Tanystropheus longobardicus (SMNS 55341); (C, F) Guchengosuchus shiguaiensis (IVPP V8808-10); (D, G) Batrachotomus kupferzellensis (SMNS 80296, [G] mirrored); (E) Aenigmastropheus parringtoni (UMZC T836); (H) Euparkeria capensis (SAM-PK-6047A); (I) Prolacerta broomi (BP/1/2675); and (J) Proterosuchus fergusi (GHG 363). Numbers indicate character-states scored in the data matrix and the arrows indicate anterior direction. Scale bars equal 2 mm in (A, G, H), and 1 cm in (B–F, I).

The absence of a notochordal canal completely piercing the vertebral centra has previously been found as an apomorphy of Archosauromorpha (e.g., Rieppel, 1989b; Ezcurra, Scheyer & Butler, 2014). Indeed, the presence of notochordal vertebrae is present in the non-saurian diapsids and lepidosauromorphs scored here (e.g., Youngina capensis: BP/1/3859; Gephyrosaurus bridensis: Evans, 1981). Conversely, choristoderans (e.g., Cteniogenys sp. Evans, 1991; Simoedosaurus lemoinei: Sigogneau-Russell, 1981) and all archosauromorphs sampled here, with the exception of Aenigmastropheus parringtoni (Ezcurra, Scheyer & Butler, 2014), lack a persistent notochordal canal (Ezcurra, Scheyer & Butler, 2014).

311. Cervical and dorsal vertebrae, anteroposterior compression of centra in the cervico-dorsal transition (=pectoral centra): moderate (0); very strong, being considerably anteroposteriorly shorter than tall (1) (New character) (Fig. 32).

Figure 32 Cervico-dorsal and dorsal vertebrae of Early and Middle Triassic non-eucrocopodan archosauriforms in (A, B) left lateral and (C) posterior views.

(A) Middle dorsal vertebra of “Chasmatosaurus” yuani (IVPP V2719); (B) anterior dorsal vertebra of Erythrosuchus africanus (NHMUK PV R3592); and (C) anterior dorsal vertebra of Sarmatosuchus otschevi (PIN 2865/68-20). Numbers indicate character-states scored in the data matrix and the arrows indicate anterior direction. Scale bars equal 2 mm in (A), 2 cm in (B), and 5 mm in (C).

The centra of the vertebrae of the cervico-dorsal transition are as long as tall or anteroposteriorly longer in the vast majority of diapsids. However, an unusual condition occurs in the erythrosuchids Erythrosuchus africanus (NHMUK PV R3592), Shansisuchus shansisuchus (Young, 1964) and Chalishevia cothurnata (PIN 4165/18), and the phytosaur Smilosuchus greogorii (USNM 18313), in which the pectoral vertebrae are strongly anteroposteriorly compressed and the centra are considerably taller dorsoventrally than long anteroposteriorly.

312. Cervical and dorsal vertebrae, neurocentral sutures: close in adults (0); remain open in sub-adults and adults (1) (Evans, 1990: 17).

The retention of open neurocentral sutures in the vertebrae of mature individuals is a feature generally associated with diapsids with aquatic habits, such as ichthyopterygians and sauropterygians (Romer, 1956). In the present taxonomic sample, the neurocentral sutures remain open throughout ontogeny only in choristoderans (Evans, 1990).

313. Cervical and dorsal vertebrae, at least one or more cervical or anterior dorsal with parallelogram-shaped centra in lateral view, in which the anterior articular surface is situated higher than the posterior one: absent (0); present (1) (Bonaparte, 1975; Sereno, 1991; Novas, 1996; Ezcurra, Lecuona & Martinelli, 2010: 115; reworded from Ezcurra, Scheyer & Butler, 2014: 174) (Figs. 11 and 33).

Figure 33 Middle and posterior cervical vertebrae of Permo-Triassic archosauromorphs in right lateral view.

(A) Seventh cervical vertebra of Protorosaurus speneri (BSPG 1995 I 5 [cast of WMsN P47361]); (B) twelfth cervical vertebra of Tanystropheus longobardicus (SMNS 54654); (C) fifth or sixth cervical vertebra of Boreopricea funerea (PIN 3708/1); and (D) seventh cervical vertebra of Prolacerta broomi (BP/1/2675). Numbers indicate character-states scored in the data matrix and the arrows indicate anterior direction. Scale bars equal 1 cm in (A), 5mm in (B), and 2 mm in (C, D).

The cervical and anterior dorsal vertebral centra of most Permo-Triassic archosauromorphs are parallelogram-shaped in lateral view, with an anterior articular surface placed dorsal to the posterior one (Ezcurra, Scheyer & Butler, 2014). By contrast, the anterior and posterior surfaces of the cervical and anterior dorsal centra of non-archosauromorph diapsids and a few archosauriforms (Proterochampsa barrionuevoi: Dilkes & Arcucci, 2012; Riojasuchus tenuisceps: PVL 3827) are placed at the main dorsoventral level in lateral view. As a result, the neck is mostly straight in the latter taxa, rather than sigmoidal as in most archosauromorphs.

314. Cervical and dorsal vertebrae, one or more vertebrae with an accessory rib articular facet between the diapophysis and parapophysis in the cervico-dorsal transition: absent (0); present (1) (Parrish, 1992: 21; Pritchard et al., 2015: 122, in part) (Figs. 10 and 31).

The articulation between the vertebrae and their respective ribs in the cervico-dorsal transition is by a pair of articular facets (parapophysis and diapophysis in the vertebra) in most archosauromorphs. An accessory articular facet placed between the diapophysis and parapophysis and on a thick anterior centrodiapophyseal lamina is present in Prolacerta broomi (BP/1/2675), proterosuchians (Proterosuchus fergusi: SAM-PK-11208; Proterosuchus alexanderi: NMQR 1484; Vonhuenia fredericki: PIN 1025/11, 419; Chasmatosuchus rossicus: PIN 2252/381; Sarmatosuchus otschevi: PIN 2865/68; Guchengosuchus shiguaiensis: IVPP V8808; Cuyosuchus huenei: MCNAM PV 2669; Garjainia prima: Huene, 1960; Garjainia madiba: Gower et al., 2014; Erythrosuchus africanus: Gower, 2003), the enigmatic archosauriform Youngosuchus sinensis (IVPP V3239), and some pseudosuchians (e.g., Batrachotomus kupferzellensis: Gower & Schoch, 2009).

315. Cervical and dorsal vertebrae, anterior centrodiapophyseal or paradiapophyseal lamina in posterior cervicals or anterior dorsals: absent (0); present (1) (Galton, 1990; Ezcurra, Scheyer & Butler, 2014: 180) (Figs. 31, 32 and 34).

Figure 34 Dorsal vertebrae of Permo-Triassic neodiapsids in (A–E) lateral and (F) ventrolateral views.

(A) Anterior-middle dorsal vertebra of Youngina capensis (BP/1/3859); (B) anterior dorsal vertebra of Aenigmastropheus parringtoni (UMZC T836); (C) anterior-middle dorsal vertebra of Tarjadia ruthae (PULR 63, mirrored); (D) first dorsal vertebra of Protorosaurus speneri (BSPG 1995 I 5 [cast of WMsN P47361]); (E) middle dorsal vertebra of Erythrosuchus africanus (NHMUK PV R3592); and (F) middle dorsal vertebrae of Chanaresuchus bonapartei (MCZ 4037). Numbers indicate character-states scored in the data matrix and the arrows indicate anterior direction. Scale bars equal 2 mm in (A), 5 mm in (B, D, F), 1 cm in (C), and 5 cm in (E).

The anterior centrodiapophyseal and paradiapophyseal laminae extend from the base of the diapophysis to the anteroventral corner of the base of the neural arch or to the anterodorsal corner of the centrum (Wilson, 1999). In the case of the paradiapophyseal lamina, this structure connects the diapophysis with the parapophysis, but it is here considered homologous to the anterior centrodiapophyseal lamina because their difference is a result of the position of the parapophysis rather than in the morphology of the laminae theirselves. Anterior centrodiapophyseal or paradiapophyseal laminae are present in most archosauromorphs scored here, but absent in non-saurian diapsids (e.g., Petrolacosaurus kansensis: Reisz, 1981; Youngina capensis: BP/1/3859), lepidosauromorphs (e.g., Gephyrosaurus bridensis: Evans, 1981; Planocephalosaurus robinsonae: Fraser & Walkden, 1984), and choristoderans (e.g., Cteniogenys sp.: Evans, 1991; Simoedosaurus lemoinei: Sigogneau-Russell, 1981). In particular, within Archosauromorpha, there are some lineages that consistently lack anterior centrodiapophyseal or paradiapophyseal laminae, such as rhynchosaurs (e.g., Mesosuchus browni: SAM-PK-6046, 7416; Rhynchosaurus articeps: Benton, 1990) and ornithischian dinosaurs (e.g., Heterodontosaurus tucki: SAM-PK-K1332).

316. Cervical and dorsal vertebrae, posterior centrodiapophyseal lamina in cervicals and/or anterior dorsals: absent (0); present (1) (Galton, 1990; Ezcurra, Scheyer & Butler, 2014: 181) (Figs. 31 and 34).

The posterior centrodiapophyseal lamina extends from the base of the diapophysis to the posterodorsal corner of the centrum or posteroventral corner of the base of the neural arch (Wilson, 1999). The distribution of this lamina among basal diapsids is similar to that of the anterior centrodiapophyseal/paradiapophyseal laminae. However, Prolacerta broomi (BP/1/2675), Boreopricea funerea (PIN 3708/1), Tasmaniosaurus triassicus (UTGD 54655), and proterosuchids (e.g., Proterosuchus alexanderi: NMQR 1484; “Chasmatosaurus” yuani: IVPP V2719) lack posterior centrodiapophyseal laminae, but possess anterior centrodiapophyseal/paradiapophyseal laminae.

317. Cervical and dorsal vertebrae, prezygodiapophyseal lamina in posterior cervicals and/or anterior dorsals: absent (0); present (1) (Bonaparte, 1986; Ezcurra, Scheyer & Butler, 2014: 182) (Fig. 34).

The prezygodiapophyseal lamina connects the base of the diapophysis with the lateral margin of the prezygapophysis (Wilson, 1999). This lamina is absent in non-archosauromorph diapsids (Ezcurra, Scheyer & Butler, 2014), rhynchosaurs (e.g., Mesosuchus browni: SAM-PK-6046, 7416; Rhynchosaurus articeps: Benton, 1990), Boreopricea funerea (PIN 3708/1), proterosuchids (e.g., Proterosuchus alexanderi: NMQR 1484; “Chasmatosaurus” yuani: IVPP V2719), Chalishevia cothurnata (PIN 4165/18), Vancleavea campi (Nesbitt et al., 2009), proterochampsids (e.g., Tropidosuchus romeri: PVL 4601; Gualosuchus reigi: PVL 4576; Chanaresuchus bonapartei: MCZ 4037), doswelliids (e.g., Doswellia kaltenbachi: USNM 244214; Tarjadia ruthae: PULR 063), and some archosaurs (e.g., Heterodontosaurus tucki: SAM-PK-K1332).

318. Cervical and dorsal vertebrae, postzygodiapophyseal lamina in posterior cervicals and/or anterior dorsals: absent (0); present (1) (Coria & Salgado, 2000; Ezcurra, Scheyer & Butler, 2014: 183) (Fig. 34).

The prezygodiapophyseal lamina connects the base of the diapophysis with the lateral margin of the postzygapophysis (Wilson, 1999). The distribution of this lamina among the sampled diapsids is very similar to that of the prezygodiapophyseal laminae, but the postzygodiapophyseal lamina is absent in the following archosauromorphs that possess the former lamina: Aenigmastropheus parringtoni (Ezcurra, Scheyer & Butler, 2014), Macrocnemus bassanii (PIMUZ T4822), Eorasaurus olsoni (Ezcurra, Scheyer & Butler, 2014), Prolacerta broomi (BP/1/2675), Tasmaniosaurus triassicus (Ezcurra, 2014), and Garjainia madiba (Gower et al., 2014).

319. Cervical and dorsal vertebrae, thick, anteroventrally-to-posterodorsally oriented tuberosity immediately below the transverse process, but both structures are not connected with each other, in posterior cervicals and anterior dorsals: absent (0); present (1) (New character) (Fig. 32).

The lateral surface of the neural arch, immediately below the base of the diapophysis is smooth in the vast majority of diapsids. Some proterosuchians possess a thick, anteroventrally-to-posterodorsally oriented tuberosity that is placed below the diapophysis and partially subdivides the centrodiapophyseal fossa, namely “Chasmatosaurus” yuani (IVPP V2719), “Gamosaurus lozovskii” (PIN 3361/14, 214), Sarmatosuchus otschevi (PIN 2865/68), and Garjainia prima (PIN 2394/5).

320. Cervical and dorsal vertebrae, gradual transverse expansion of the distal half of the neural spine: absent (0); present, but lacking distinct mammillary processes on the lateral surface of the neural spine (1); present, with distinct mammillary processes on the lateral surface of the neural spine (2) (modified from Laurin, 1991: H3; Reisz, Laurin & Marjanović, 2010: 76; Ezcurra, Scheyer & Butler, 2014: 75; Pritchard et al., 2015: 125, in part) (Figs. 31, 32 and 34).

The distal half of the neural spine of diapsids exhibits different kinds and degrees of transverse expansion. The presence of a transverse expansion restricted to the distal margin of the neural spine has been historically called spine table in both pseudosuchians (see Nesbitt, 2011) and avemetatarsalians (e.g., Herrerasaurus ischigualastensis: Novas, 1993). Contrasting with the condition in archosaurs (including phytosaurs), multiple diapsid clades possess a gradual transverse expansion of the distal half of the neural spine, but lacking a distinct, distally restricted transverse expansion that forms a T-shaped distal end in anterior or posterior view. As a result, it is assumed here that the condition in these non-archosaurian diapsids is not homologous to the spine table of archosaurs (contra Nesbitt, 2011).

The early neodiapsids Petrolacosaurus kansensis (Reisz, 1981) and Araeoscelis gracilis (Vaughn, 1955) possess a pair of transverse expansions on the distal half of the neural spine that they do not reach its anterior and posterior margins. These transverse expansions present in araeoscelidians are called mammillary processes, and Ezcurra & Butler (2015a) proposed to use this name for topologically very similar structures present in Proterosuchus species from South Africa. These mammillary processes are also present in Protorosaurus speneri (BSPG 1995 I 5, cast of WMsN P47361), Prolacerta broomi (BP/1/2675), Boreopricea funerea (PIN 3708/1), “Chasmatosaurus” yuani (IVPP V4067) and Chasmatosuchus rossicus (PIN 3200/217). In other diapsids, including Simoedosaurus lemoinei (MNHN.F.BR12208), Pamelaria dolichotrachela (ISI 316/1), Vonhuenia fredericki (PIN 1025/11), the “proterosuchid from Long Reef” (SAM P41754), Sarmatosuchus otschevi (PIN 2865/68), Cuyosuchus huenei (MCNAM PV 2669), erythrosuchids (e.g., Guchengosuchus shiguaiensis: IVPP V8808; Garjainia prima: PIN 2394/5; Shansisuchus shansisuchus: Wang et al., 2013), Euparkeria capensis (SAM-PK-5867), and Doswellia kaltenbachi (USNM 244214), the distal half of the neural spine expands gradually transversely, but this expansion is homogenously developed anteroposteriorly (i.e., they lack mammillary processes).

321. Cervical and dorsal vertebrae, spine table on the distal end of the postaxial neural spines (not mammillary process): absent (0); present in cervicals, but not in dorsals (1); present in dorsal, but not in cervicals (2); present in both cervicals and dorsals (3) (modified from Dilkes, 1998; Ezcurra, Lecuona & Martinelli, 2010: 49, in part; Nesbitt, 2011: 191 and 197, in part; Dilkes & Arcucci, 2012: 53 and 54; Pritchard et al., 2015: 125, in part) (Figs. 31 and 32).

The spine table is a transverse expansion restricted to the distal end of the neural spine, which gives the spine a T-shaped contour in anterior or posterior view. The spine tables are restricted to phytosaurs, ornithosuchids, suchians, and Herrerasaurus ischigualastensis (see comments for character 320). Spine tables are present on both cervical and dorsal vertebrae in phytosaurs (Parasuchus hislopi: ISI R42; Smilosuchus gregorii: USNM 18313), ornithosuchids (Ornithosuchus longidens: Walker, 1964; Riojasuchus tenuisceps: PVL 3827), Nundasuchus songeaensis (Nesbitt et al., 2014), gracilisuchids (e.g., Turfanosuchus dabanensis: IVPP V3237; Gracilisuchus stipanicicorum: MCZ 4118), and loricatans (e.g., Batrachotomus kupferzellensis: Gower & Schoch, 2009; Prestosuchus chiniquensis: UFRGS-PV-0152-T). By contrast, the spine tables are present on the dorsal vertebrae, but not on the cervical vertebrae, of Herrerasaurus ischigualastensis (PVL 2566, PVSJ 407) and Aetosauroides scagliai (MCP 13a-b-PV; PVL 2059, 2073).

322. Cervical and dorsal vertebrae, distal surface of transverse expansion of the neural spine: convex (0); approximately flat (1) (modified from Dilkes, 1998; Ezcurra, Lecuona & Martinelli, 2010: 49, in part; Nesbitt, 2011: 197, in part). This character is inapplicable in taxa that lack a transverse expansion of the distal end of the neural spine or possess mammillary processes (Fig. 31).

See comments in Nesbitt (2011: character 197).

323. Cervical and dorsal vertebrae, outline of the spine tables in dorsal view: suboval or subrectangular (0); subtriangular or heart-shaped (1) (Gauthier, 1984; modified from Nesbitt, 2011: 191). Character inapplicable in taxa that lack spine tables (Fig. 28 of Nesbitt, 2011).

See comments in Nesbitt (2011: character 191).

324. Cervical vertebrae, number of vertebrae in the neck: fewer than eight (0); eight to ten (1); more than ten (2) (modified from Gauthier, 1986), ORDERED.

Petrolacosaurus kansensis (Reisz, 1981), Youngina capensis (Smith & Evans, 1996), Protorosaurus speneri (BSPG 1995 I 5, cast of WMsN P47361), and Trilophosaurus buettneri (Spielmann et al., 2008) possess fewer than eight cervical vertebrae, whereas other species sampled here possess eight or more cervicals. Among the latter taxa, Tanystropheus longobardicus represents an exception with 13 cervical vertebrae (Wild, 1973; Nosotti, 2007).

325. Cervical vertebrae, atlantal articulation facet on the axial intercentrum: saddle-shaped (0); concave with upturned lateral borders (1) (Gauthier, 1986; Nesbitt, 2011: 178) (Fig. 28 of Nesbitt, 2011).

See comments in Nesbitt (2011: character 178).

326. Cervical vertebrae, centrum of atlas in mature individuals: separate from axial intercentrum (0); fused to axial intercentrum (1) (Nesbitt et al., 2015: 243) (Fig. 30).

See comments in Nesbitt et al. (2015: character 243).

327. Cervical vertebrae, ventral surface of the centrum on anterior cervicals: transversely convex (0); with a low median longitudinal keel (1); with a median longitudinal keel that extends ventral to the centrum rim in at least one anterior cervical (2) (modified from Nesbitt, 2011: 190; Pritchard et al., 2015: 108, in part), ORDERED (Fig. 11).

See comments in Nesbitt (2011: character 190).

328. Cervical vertebrae, height of neural spine of the axis: dorsoventrally tall (0); strongly dorsoventrally short (1) (New character) (Fig. 30).

The neural spine of the axis is dorsoventrally taller than the height of its respective centrum in the vast majority of diapsids. However, in tanystropheids (e.g., Tanystropheus longobardicus: PIMUZ T2189; Amotosaurus rotfeldensis: SMNS 50830; Macrocnemus bassanii: PIMUZ T4822), Prolacerta broomi (BP/1/2675), Proterochampsa barrionuevoi (Dilkes & Arcucci, 2012), Doswellia kaltenbachi (USNM 244214), and some specimens of Silesaurus opolensis (ZPAL AbIII/361) the neural spine of the axis is dorsoventrally low, being lower than the height of the axial centrum.

329. Cervical vertebrae, shape of the neural spine of the axis: expanded posterodorsally or the height of the anterior portion is equivalent to the posterior height (0); expanded anterodorsally (1) (Nesbitt, 2011: cf. 179; Nesbitt et al., 2015: 244) (Fig. 30).

See comments in Nesbitt (2011: character 179) and Nesbitt et al. (2015: character 244).

330. Cervical vertebrae, dorsal margin of the neural spine of the axis: mostly dorsally convex (0); mostly straight or dorsally concave (1) (Makovicky & Sues, 1998) (Fig. 30).

331. Cervical vertebrae, lengths of the fourth or fifth cervical centra versus the heights of their anterior articular surfaces: 0.63–2.67 (0); 2.92–4.12 (1); 6.09–6.80 (2); 14.16–14.33 (3) (Laurin, 1991: H1; modified from Senter, 2004: 28, Müller, 2004: 174, and Reisz, Laurin & Marjanović, 2010: 69; Ezcurra, Lecuona & Martinelli, 2010: 47 and 103, in part; Nesbitt, 2011: 181 and 183, in part; cf. Dilkes & Arcucci, 2012: 51; modified from Ezcurra, Scheyer & Butler, 2014: 68, and Nesbitt et al., 2015: 214), ORDERED (Fig. 15).

See comments in Nesbitt (2011: characters 181 and 183).

332. Cervical vertebrae, diapophysis and parapophysis of anterior to middle cervical postaxial vertebrae: single facet or both situated on the same process (0); situated on different processes and well separated (1); situated on different processes and nearly touching (2) (modified from Nesbitt, 2011: 184, and Pritchard et al., 2015: 110, 112) (Fig. 30).

See comments in Nesbitt (2011: character 184).

333. Cervical vertebrae, position of diapophysis or dorsal margin of synapophysis in anterior postaxial cervicals: at or near dorsoventral level of pedicles (0); near the dorsoventral mid-point of the centrum (1) (Gauthier, 1984; Benton, 1985; Pritchard et al., 2015: 111).

The diapophysis or the dorsal margin of the synapophysis for the cervical rib is placed at the base of the neural arch in non-archosauromorph diapsids (e.g., Petrolacosaurus kansensis: Reisz, 1981; Youngina capensis: BP/1/3859; Planocephalosaurus robinsonae: Fraser & Walkden, 1984; Cteniogenys sp.: Evans, 1991; Simoedosaurus lemoinei: MNHN.F.BL9425). By contrast, in the vast majority of archosauromorphs the diapophysis or synapophysis is placed at mid-height on the anterior margin of the lateral surface of the centrum. Exceptions to the latter condition among archosauromorphs are observed in Euparkeria capensis (SAM-PK-5867), Aetosauroides scagliai (PVL 2059, 2073), and Heterodontosaurus tucki (SAM-PK-K1332).

334. Cervical vertebrae, longitudinal lamina or tuberosity extended posteriorly from the base of the transverse process in postaxial anterior and middle cervicals: absent or poorly developed, not well laterally developed (0); strongly developed, flaring laterally as a prominent and thick, wing-like shelf (1) (New character) (Fig. 11).

The presence of a longitudinal tuberosity that extends posteriorly from the base of the diapophysis to the posterodorsal corner of the centrum is a common condition among archosauromorphs. In particular, this tuberosity is strongly laterally flared and broadly visible as a shelf in ventral view in Chasmatosuchus rossicus (PIN 3200/217, 472), Chasmatosuchus magnus (PIN 951/65), and “Gamosaurus lozovskii” (PIN 3361/13, 94, 183).

335. Cervical vertebrae, posterior portion of the neural arch ventral to the postzygapophysis in postaxial cervicals: flat (0); with a shallow, posterolaterally facing fossa (1) (New character) (Fig. 11).

The lateral surface of the neural arch immediately ventral to the articular facet of the postzygapophysis is flat in the vast majority of diapsids. Some proterosuchians (e.g., some specimens of Proterosuchus fergusi: BP/1/3993, and Chasmatosuchus rossicus: PIN 3200/217; Garjainia prima: PIN 2394/5; Erythrosuchus africanus: NHMUK PV R3592) and Gracilisuchus stipanicicorum (PULR 08) possess a shallow, posterolaterally facing fossa in this area of the neural arch.

336. Cervical vertebrae, epipophysis in postaxial cervicals: absent (0); present in at least the third to fifth cervical vertebrae (1) (Gauthier, 1986; Sereno et al., 1993; modified from Nesbitt, 2011: 186, 187; Pritchard et al., 2015: 119) (Figs. 30 and 33).

See comments in Nesbitt (2011: characters 186, 187).

337. Cervical vertebrae, excavation immediately lateral to the base of postaxial cervical neural spines: absent (0); shallow (1); represented by a deep pocket or pit (2) (modified from Reisz & Dilkes, 2003: 47; Reisz, Laurin & Marjanović, 2010: 71; Ezcurra, Scheyer & Butler, 2014: 70), ORDERED (Figs. 11, 33 and 34).

An excavation immediately lateral to the base of the neural spine, being placed approximately at mid-length between the bases of the zygapophyses, is present in multiple diapsid species. This excavation is shallow and not very well defined in most diapsids that possess this feature, such as Petrolacosaurus kansensis (Reisz, 1981), Protorosaurus speneri (BSPG 1995 I 5, cast of WMsN P47361), Eorasaurus olsoni (PIN 156/108-110), Proterosuchus alexanderi (NMQR 1484), Vonhuenia fredericki (PIN 1025/11), Guchengosuchus shiguaiensis (IVPP V8808), Cuyosuchus huenei (MCNAM PV 2669) and Euparkeria capensis (SAM-PK-5867). By contrast, this excavation is deep and well defined as a subcircular pocket in Chasmatosuchus rossicus (PIN 3200/381, 217), Chasmatosuchus magnus (PIN 951/65) and “Gamosaurus lozovskii” (PIN 3361/13, 14), some of the cervical vertebrae of Proterosuchus fergusi (BP/1/3993), and Gracilisuchus stipanicicorum (PULR 08).

338. Cervical vertebrae, anterior cervical vertebrae (presacral vertebrae 3–5) postzygapophyses: separated posteriorly (0); connected through a horizontal lamina (= transpostzygapophyseal lamina) with a notch at the midline (1) (Nesbitt et al., 2015: 213).

See comments in Nesbitt et al. (2015: character 213).

339. Cervical vertebrae, shape of the postaxial neural spines in lateral view: sub-triangular (0); rectangular (1) (Laurin, 1991: C1; Reisz, Laurin & Marjanović, 2010: 72; Ezcurra, Scheyer & Butler, 2014: 71).

The cervical neural spines of most diapsids possess subparallel anterior and posterior margins in lateral view, which form a subrectangular neural spine. By contrast, in some lepidosauromorphs the anterior and posterior margins of the cervical neural spines converge at the distal tip of the spine in an obtuse angle, forming a subtriangular neural spine in lateral view (e.g., Gephyrosaurus bridensis: Evans, 1981).

340. Cervical vertebrae, distinct longitudinal lamina extending along the lateral surface of the centrum at mid-height in anterior and middle postaxial cervical vertebrae: absent (0); present (1) (Ezcurra, Scheyer & Butler, 2014: 169) (Fig. 30).

The ventrolateral surface of the centrum is smoothly dorsoventrally convex in most diapsids. In the tanystropheids Macrocnemus bassanii (PIMUZ T4822) and Tanystropheus longobardicus (PIMUZ T2818), Eorasaurus olsoni (PIN 156/109), the erythrosuchids Garjainia prima (PIN 2394/5) and Garjainia madiba (BP/1/5360), and Doswellia kaltenbachi (USNM 244214) there is a longitudinal lamina or tuberosity that extends posteriorly from the base of the parapophysis and forms a sharp change in slope between the ventral and lateral surfaces of the centrum.

341. Cervical vertebrae, longitudinal lamina connecting the prezygapophysis and postzygapophysis on the third cervical neural arch: absent (0); present (1) (Ezcurra, Scheyer & Butler, 2014: 170).

A sharp longitudinal lamina connects the dorsolateral edges of the prezygapophysis and postzygapophysis in the third cervical vertebra of Macrocnemus bassanii (PIMUZ T4822), Prolacerta broomi (BP/1/2675) and Aetosauroides scagliai (PVL 2059). In other diapsids, the surface between the base of the diapophysis and the neural spine is smooth and not interrupted by a longitudinal lamina.

342. Cervical vertebrae, shape of postaxial anterior cervical neural spines: tall, with height and length approximately equal or height larger (0); long and low, with height lower than length (1) (Dilkes, 1998: 82; Ezcurra, Scheyer & Butler, 2014: 171; Pritchard et al., 2015: cf. 113) (Fig. 11).

The postaxial cervical neural spines are dorsoventrally low, being lower than the anteroposteiror length of the spine at its base, in Petrolacosaurus kansensis (Reisz, 1981), Protorosaurus speneri (BSPG 1995 I 5, cast of WMsN P47361), tanystropheids (e.g., Amotosaurus rotfeldensis: SMNS 50830; Macrocnemus bassanii: PIMUZ T4822), Pamelaria dolichotrachela (ISI R316/1), Azendohsaurus madagaskarensis (Nesbitt et al., 2015), Prolacerta broomi (BP/1/2675), doswelliids (Jaxtasuchus salomoni: SMNS 91083; Doswellia kaltenbachi: USNM 244214) and some dinosauromorphs (e.g., Marasuchus lilloensis: PVL 3872; Herrerasaurus ischigualastensis: PVSJ 407).

343. Cervical vertebrae, anterior and middle postaxial cervical neural spines with an anterior overhang: absent (0); present (1) (Senter, 2004: 30; Ezcurra, Scheyer & Butler, 2014: 172; Pritchard et al., 2015: 115) (Figs. 30 and 33).

The anterior margin of the neural spine is vertical and mostly straight in most diapsids. In Protorosaurus speneri (BSPG 1995 I 5, cast of WMsN P47361), tanystropheids (e.g., Amotosaurus rotfeldensis: SMNS 50830; Macrocnemus bassanii: PIMUZ T4822; Tanystropheus longobardicus: PIMUZ T2189), the allokotosaurians Azendohsaurus madagaskarensis (Nesbitt et al., 2015) and Trilophosaurus buettneri (Spielmann et al., 2008), Prolacerta broomi (BP/1/2675), Guchengosuchus shioguaiensis (IVPP V8808), Yarasuchus deccanensis (ISI R334), Tropidosuchus romeri (PVL 4601), doswelliids (Jaxtasuchus salomoni: SMNS 91083; Doswellia kaltenbachi: USNM 244214), and gracilisuchids (Gracilisuchus stipanicicorum: PULR 08; Turfanosuchus dabanensis: IVPP V3237) the anterior margin of the neural spine slants strongly anteriorly in lateral view and produces a distinct anterior overhang of the neural spine.

344. Cervical vertebrae, relative location of dorsal margin of anterior and middle cervical postaxial neural spines: spines are equivalent in height and length to other cervical neural spines (0); spines are dorsoventrally depressed at their anteroposterior midpoints, leaving them little more than midline dorsal ridges (1) (Pritchard et al., 2015: 118) (Fig. 30).

The dorsal margin of the neural spine of the middle cervical vertebrae is straight or gently convex in lateral view in most diapsids. By contrast, in Amotosaurus rotfeldensis (SMNS 50830) and Tanystropheus longobardicus (PIMUZ T2189) the dorsal margin of the neural spine is deeply concave and, as a result, the neural spine is partially bifurcated in lateral view (Pritchard et al., 2015).

345. Cervical vertebrae, position of the mammillary processes of the neural spines along the neck: present from the fourth presacral (0); present from the fifth presacral (1); present from the sixth or seventh presacral (2); present from the eighth or ninth presacral (3) (New character), ORDERED. Character inapplicable in taxa that lack mammillary processes.

The placement of the most anterior mammillary processes in the cervical series varies in the species that possess this feature. The mammillary processes are present from the fourth or fifth presacral vertebra in Petrolacosaurus kansensis (Reisz, 1981), Proterosuchus alexanderi (NMQR 1484) and “Chasmatosaurus” yuani (IVPP V4067), from the fifth to seventh presacral in Boreopricea funerea (PIN 3708/1) and Proterosuchus fergusi (SAM-PK-11208), and from the eighth or ninth presacral in Protorosaurus speneri (BSPG 1995 I 5, cast of WMsN P47361) and Prolacerta broomi (BP/1/2675).

346. Cervical vertebrae, postaxial cervical intercentra: present (0); absent (1) (Dilkes, 1998: 79; Müller, 2004: 43; Ezcurra, Lecuona & Martinelli, 2010: 45; Nesbitt, 2011: 177, in part; Dilkes & Arcucci, 2012: 50, in part; Ezcurra, Scheyer & Butler, 2014: 168; Pritchard et al., 2015: 106).

See comments in Nesbitt (2011: character 177).

347. Cervical and dorsal ribs, tuberculum in posterior cervical or anterior dorsal ribs: short (0); long and distinct (1) (Ezcurra, Lecuona & Martinelli, 2010: 92).

The tuberculum is short, developed as a sub-quadrangular peduncle in anterior or posterior views in non-archosauromorph diapsids (e.g., Acerosodontosaurus piveteaui: MNHN 1908-32-57) and the archosauromorphs Protorosaurus speneri (BSPG 1995 I 5, cast of WMsN P47361) and Doswellia kaltenbachi (USNM 244214). In other taxa sampled here, the tuberculum of the posterior cervical or anterior dorsal ribs is longer than tall and distinctly offset from the shaft.

348. Cervical and dorsal ribs, at least one rib of the cervico-dorsal transition with a thin lamina webbing tuberculum and capitulum: absent (0); present (1) (New character). This character is inapplicable in taxa with holocephalous ribs or poorly differentiated tuberculum and capitulum.

In the vast majority of diapsids, the tuberculum and capitulum of the cervico-dorsal ribs meet at their bases without any gradual transition. By contrast, in Eorasaurus olsoni (PIN 156/110), Prolacerta broomi (BP/1/2675), Proterosuchus alexanderi (NMQR 1484), Guchengosuchus shiguaiensis (IVPP V8808), Erythrosuchus africanus (SAM-PK-3028) and Youngosuchus sinensis (IVPP V3239) a thin lamina connects the bases of the capitulum and tuberculum and in some species supports the third articular facet of the rib. This character is considered independent from the presence of an accessory articular facet in the cervico-dorsal vertebrae of some archosauromorphs because there is no evidence of this facet in Eorasaurus olsoni (Ezcurra, Scheyer & Butler, 2014).

349. Cervical ribs, shape: short, being less than two times the length of its respective vertebra, and tapering at a high angle to the neck (0); short, being less than two times the length of its respective vertebra, and shaft parallel to the neck (1); very long, being two times the length of its respective vertebra, and parallel to the neck (2) (Gauthier, 1986; Benton & Clark, 1988; Juul, 1994; Dilkes, 1998: 77; Müller, 2004: 102; Ezcurra, Lecuona & Martinelli, 2010: 116, in part; Nesbitt, 2011: 196, in part; Ezcurra, Scheyer & Butler, 2014: 173; Pritchard et al., 2015: 104, in part) (Figs. 30 and 28 of Nesbitt, 2011).

See comments in Nesbitt (2011: character 196).

350. Cervical ribs, accessory process on anterolateral surface of anterior cervical ribs: absent (0); present (1) (Laurin, 1991: H4; Müller, 2004: 48; Reisz, Laurin & Marjanović, 2010: 77; Ezcurra, Scheyer & Butler, 2014: 76; Pritchard et al., 2015: 105) (Fig. 30).

The anterior cervical ribs of all archosauromorphs scored here, with the exception of Heterodontosaurus tucki (SAM-PK-K1332), possess an accessory, anteriorly projecting and tapering process on their anterolateral surface. This process is absent in the non-archosauromorph diapsids sampled here.

351. Dorsal vertebrae, length versus height of the centrum in anterior dorsals: 0.45–1.10 (0); 1.18–2.00 (1); 2.19–2.74 (2) (Sereno, 1999; modified from Ezcurra, Scheyer & Butler, 2014: 176), ORDERED.

352. Dorsal vertebrae, length versus height of the centrum in posterior dorsals: 0.66–1.39 (0); 1.48–1.86 (1); 1.95–2.04 (2); 2.39–2.46 (3) (modified from Ezcurra, Lecuona & Martinelli, 2010: 87), ORDERED.

Characters 351 and 352 describe the proportional anteroposterior elongation of the anterior and posterior dorsal centra, respectively, with respect to the dorsoventral height of their respective centrum (measured at either the anterior or posterior articular surface of the centrum).

353. Dorsal vertebrae, ventral surface of middle and posterior centra: transversely convex (0); ridged, with slightly swollen sides (1); single keel (2); double keel (3) (Laurin, 1991: H2; Reisz & Dilkes, 2003: 44; Reisz, Laurin & Marjanović, 2010: 74; Ezcurra, Scheyer & Butler, 2014: 73).

The ventral surface of the middle and posterior dorsal centra is transversely convex, without a longitudinal ridge or keel, in most diapsids. In Petrolacosaurus kansensis (Reisz, 1981), Tanystropheus longobardicus (SMNS 55341) and the “proterosuchid from Long Reef” (SAM P41754) there is a single, median longitudinal keel on the ventral surface of the centrum of the middle and posterior dorsal vertebrae, whereas this keel is restricted to only some vertebrae of this region of the trunk in Erythrosuchus africanus (NHMUK PV R3592). A pair of longitudinal ventral keels is present in Vancleavea campi (Parker & Barton, 2008), whereas some middle to posterior dorsal vertebrae of Jaxtasuchus salomoni possess a single ventral keel and others a double keel (SMNS 91352A, B). The archosaur Nundasuchus songeaensis lacks a ventral keel on some middle to posterior dorsal vertebrae, whereas there is a double keel in others (Nesbitt et al., 2014). The species that possess a combination of character-states in the middle-posterior region of the trunk were scored as polymorphic.

354. Dorsal vertebrae, lateral fossa on the centrum below the neurocentral suture: absent (0); present, but not well-rimmed (1); present and well-rimmed (2) (Gauthier, 1986; modified from Ezcurra, Lecuona & Martinelli, 2010: 88),ORDERED (Figs. 31 and 34).

In the vast majority of non-archosauriform diapsids, the lateral surface of the dorsal centra is dorsoventrally convex or flat, but in Protorosaurus speneri (BSPG 1995 I 5, cast of WMsN P47361) and some dorsal vertebrae of Jesairosaurus lehmani (ZAR 11) and Pamelaria dolichotrachela (ISI R316) there is a shallow, not well-rimmed oval fossa on the lateral surface of the centrum. The latter condition is present in most archosauriforms, with the exception of Sarmatosuchus otschevi (PIN 2865/68), Proterochampsa barrionuevoi (Dilkes & Arcucci, 2012), Archeopelta arborensis (CPEZ-239a), and Riojasuchus tenuisceps (PVL 3827). In particular, the fossa on the lateral surface of the dorsal centra is deeper and well-rimmed in Koilamasuchus gonzalezdiazi (MACN-Pv 18119), “Dongusia colorata” (PIN 268/2), and suchian archosaurs (e.g., Turfanosuchus dabanensis: IVPP V3237; Aetosauroides scagliai: MCP 13a-b-PV, PVL 2073; Batrachotomus kupferzellensis: SMNS 80296, 80300, 80305, 80309).

355. Dorsal vertebrae, subcentral foramen on the lateral surface of the centra: absent (0); present (1) (Ezcurra, Scheyer & Butler, 2014: 177) (Fig. 34).

The subcentral foramen is a moderately large, circular opening that pierces the centre of the lateral surface of the centrum. This foramen is present in Acerosodontosaurus piveteaui (MNHN 1908-32-57), Gephyrosaurus bridensis (Evans, 1981), the choristoderans Cteniogenys sp. (Evans, 1991) and Simoedosaurus lemoinei (MNHN.F.BR2075; Sigogneau-Russell, 1981), and the early archosauromorph Aenigmastropheus parringtoni (Ezcurra, Scheyer & Butler, 2014).

356. Dorsal vertebrae, diapophysis and parapophysis in anterior dorsals: close to the body of the midline (0); expanded on stalks (1) (Nesbitt, 2011: 199).

See comments in Nesbitt (2011: character 199).

357. Dorsal vertebrae, ratio between transverse width of diapophysis and length of the centrum in anterior dorsals: <0.70 (0); >0.75 (1) (Ezcurra, Scheyer & Butler, 2014: 178) (Fig. 31).

The diapophyses of the anterior dorsal vertebrae of most non-archosauriform archosauromorphs project slightly laterally, with a transverse length lower than 0.75 times the anteroposterior length of its respective centrum. In Pamelaria dolichotrachela (ISI R316/1), Trilophosaurus buettneri (Spielmann et al., 2008), Eorasaurus olsoni (Ezcurra, Scheyer & Butler, 2014) and the vast majority of archosauriforms (e.g., Proterosuchus alexanderi: NMQR 1484; Cuyosuchus huenei: MCNAM PV 2669; Erythrosuchus africanus: NHMUK PV R3592; Chanaresuchus bonapartei: MCZ 4037; Riojasuchus tenuisceps: PVL 3827; Heterodontosaurus tucki: SAM-PK-K1332) the diapophysis is proportionally very long and exceeds 0.75 times the length of its respective centrum. In particular, some archosaurs possess short diapophysis on the anterior dorsal vertebrae, including the loricatans Batrachotomus kupferzellensis (SMNS 80296, 80309) and Prestosuchus chiniquensis (UFRGS-PV-0152-T), and the early dinosauriforms Lewisuchus admixtus (PULR 01) and Silesaurus opolensis (Piechowski & Dzik, 2010).

358. Dorsal vertebrae, development of the transverse process in middle and posterior dorsals: short (0); moderately long (1); extremely long, being considerably broader than their respective centra (2) (Laurin, 1991: F5; Reisz, Laurin & Marjanović, 2010: 75; modified from Ezcurra, Scheyer & Butler, 2014: 74) (Fig. 32).

The transverse process of the middle and posterior dorsal vertebrae are short, projecting only slightly beyond the lateral surface of the neural arch, in Petrolacosaurus kansensis (Reisz, 1981), Acerosodontosaurus piveteaui (MNHN 1908-32-57), Youngina capensis (BP/1/3859), lepidosauromorphs (e.g., Gephyrosaurus bridensis: Evans, 1991), Simoedosaurus lemoinei (MNHN.F.BR2075, BR12208) and Jesairosaurus lehmani (ZAR 13). In all other diapsids scored here, the transverse process is proportionally longer than in the species listed above, but in the doswelliids Doswellia kaltenbachi (USNM 244214) and probably Tarjadia ruthae (PULR 063) the transverse process is strongly laterally extended, being transversely broader than its respective centrum.

359. Dorsal vertebrae, hyposphene-hypantrum accessory intervertebral articulation in middle-posterior dorsals: absent (0); present (1) (Gauthier, 1986; Juul, 1994; Ezcurra, Lecuona & Martinelli, 2010: 117; modified from Nesbitt, 2011: 195).

See comments in Nesbitt (2011: character 195) (Figs. 31 and 32).

360. Dorsal vertebrae, zygosphene-zygantrum articulation: absent (0); present (1) (Rieppel, Mazin & Tchernov, 1999; Müller, 2004: 44; Ezcurra, Scheyer & Butler, 2014: 186; Pritchard et al., 2015: 130) (Fig. 31).

The zygosphene-zygantrum is an accessory intervertebral articulation present on the neural arches of lepidosauromorphs (Gauthier, Kluge & Rowe, 1988). The zygantrum is developed as a pair of ventrolaterally facing articular facets raised on a median structure that is usually confluent posteriorly with the anterior end of the neural spine. The zygosphene are dorsomedially facing articular facets placed within the postspinal fossa and medial to the postzygapophyses. These accessory articular facets are present in the early rhynchocephalians Gephyrosaurus bridensis (Evans, 1991) and Planocephalosaurus robinsonae (Fraser & Walkden, 1984) in the taxonomic sample of this analysis.

361. Dorsal vertebrae, dorsally opening pit lateral to the base of the neural spine: absent (0); shallow (1); developed as a deep pit (2) (Dilkes, 1998: 84; Müller, 2004: 103, in part; Ezcurra, Lecuona & Martinelli, 2010: 48; Dilkes & Arcucci, 2012: 52; Ezcurra, Scheyer & Butler, 2014: 184), ORDERED (Fig. 34).

A shallow pit that opens immediately lateral to the base of the neural spine in the dorsal vertebrae occurs in various diapsids, including Protorosaurus speneri (BSPG 1995 I 5, cast of WMsN P47361), Pamelaria dolichotrachela (ISI R316), Azendohsaurus madagaskarensis (Nesbitt et al., 2015), early rhynchosaurs (e.g., Howesia browni: SAM-PK-5886), Prolacerta broomi (BP/1/2675), Boreopricea funerea (PIN 3708/1), several basal archosauriforms (e.g., Proterosuchus alexanderi: NMQR 1484; Cuyosuchus huenei: MCNAM PV 2669; Garjainia prima: PIN 2394/5; Euparkeria capensis: UMCZ T692), gracilisuchids (e.g., Turfanosuchus dabanensis: IVPP V3237), and Batrachotomus kupferzelensis (SMNS 80296, 80300, 80305). A deeper pit in the same position as in the aforementioned taxa is present in Petrolacosaurus kansensis (Reisz, 1981), the Long Reef proterosuchid (SAM P41754), Guchengosuchus shiguaiensis (IVPP V8808), and Erythrosuchus africanus (NHMUK PV R3592).

362. Dorsal vertebrae, anterior and middle dorsal neural spines: subrectangular, with the anterior margin vertical, anterodorsally or slightly posterodorsally inclined (0); subtriangular, with the anterior margin strongly posterodorsally oriented (1) (Ezcurra, Scheyer & Butler, 2014: 179).

The neural spines of the anterior and middle dorsal vertebrae of the lepidosauromorphs Gephyrosaurus bridensis (Evans, 1991) and Planocephalosaurus robinsonae (Fraser & Walkden, 1984) are subtriangular in lateral view, with a dorsally oriented apex. The same condition occurs in some of the middle dorsal vertebrae of Petrolacosaurus kansensis, but the anterior dorsals possess subrectangular neural spines in lateral view (Reisz, 1981). In all other diapsids scored here, the neural spines of the anterior and middle dorsal vertebrae are subrectangular in lateral view.

363. Dorsal vertebrae, fan-shaped neural spine in lateral view: absent (0); present (1) (New character).

The dorsal vertebrae of tanystropheids (e.g., Amotosaurus rotfeldensis: SMNS 54783; Macrocnemus bassanii: PIMUZ T4822; Tanystropheus longobardicus: Nosotti, 2007), Tropidosuchus romeri (PVL 4601), gracilisuchids (e.g., Turfanosuchus dabanensis: IVPP V3237; Gracilisuchus stipanicicorum: PULR 08), and some basal dinosauriforms (e.g., Marasuchus lilloensis: PVL 3870; Lewisuchus admixtus: PULR 01) possess fan-shaped neural spines, being trapezoidal in lateral view and anteroposteriorly longer at their distal margin than at their base. By contrast, the dorsal neural spines of other diapsids are not expanded anteriorly and posteriorly toward their distal ends and are subrectangular in lateral view.

364. Dorsal vertebrae, position of middle dorsal neural spines: situated at mid-length between the zygapophyses (0); displaced posteriorly from mid-length between the zygapophyses (1) (Ezcurra, Lecuona & Martinelli, 2010: 90; Dilkes & Arcucci, 2012: 55).

The neural spine of the middle dorsal vertebrae of the vast majority of diapsids is displaced posteriorly from the point of mid-length between the zygapophyses in lateral view. However, the neural spine is centred between the zygapophyses in Doswellia kaltenbachi (USNM 244214).

365. Dorsal vertebrae, position of the mammillary processes of the neural spines in the trunk: extend up to the tenth presacral (0); extend up to the eleventh presacral (1); extend up to the twelfth presacral (2); extend up to the thirteenth presacral (3); extend up to the sixteenth presacral or beyond (4) (New character), ORDERED. This character is inapplicable in taxa that lack mammillary processes.

The position of the most posterior mammillary processes in the dorsal series varies in different species. The mammillary processes extend up to the tenth presacral vertebra in Protorosaurus speneri (BSPG 1995 I 5, cast of WMsN P47361) and some specimens of Proterosuchus fergusi (SAM-PK-11208), the eleventh presacral in Petrolacosaurus kansensis (Reisz, 1981), the twelfth presacral in Prolacerta broomi (BP/1/2675) and some specimens of Proterosuchus fergusi (GHG 363), the thirteenth presacral in “Chasmatosaurus” yuani (IVPP V4067), and the sixteenth presacral or beyond in Proterosuchus alexanderi (NMQR 1484). The condition in the early archosauromorph Boreopricea funerea is not well constrained, but the mammillary processes extend at least up to the twelfth presacral vertebra.

366. Dorsal vertebrae, intercentra: present (0); absent (1) (Rieppel, Mazin & Tchernov, 1999; Müller, 2004: 42; Ezcurra, Lecuona & Martinelli, 2010: 46; Dilkes & Arcucci, 2012: 50, in part; Ezcurra, Scheyer & Butler, 2014: 175; Pritchard et al., 2015: 128).

Intercentra in the dorsal series are absent in the following taxa sampled here: Protorosaurus speneri (BSPG 1995 I 5, cast of WMsN P47361), Macrocnemus bassanii (PIMUZ T2472, T4355, T4822), Tanystropheus longobardicus (Wild, 1973; Nosotti, 2007), Jesairosaurus lehmani (Jalil, 1997), Azendohsaurus madagaskarensis (Nesbitt et al., 2015), Rhynchosaurus articeps (Benton, 1990),“Chasmatosaurus” yuani (IVPP V4067), Yarasuchus deccanensis (ISI R334), proterochampsids (e.g., Proterochampsa barrionuevoi: Dilkes & Arcucci, 2012; Tropidosuchus romeri: PVL 4601; Gualosuchus reigi: PVL 4576); doswelliids (e.g., Archeopelta arborensis: CPEZ-239a; Jaxtasuchus salomoni: SMNS 91352A, B; Doswellia kaltenbachi: USNM 244214), phytosaurs (e.g., Nicrosaurus kapffi: Huene, 1923),ornithosuchids (e.g., Riojasuchus tenuisceps: PVL 3827), suchians (e.g., Gracilisuchus sitpanicicorum: PULR 08; Batrachotomus kupferzellensis: Gower & Schoch, 2009), and avemetatarsalians (e.g., Marasuchus lilloensis: PVL 3870; Lewisuchus admixtus: PULR 01; Herrerasaurus ischigualastensis: PVL 2566).

367. Dorsal ribs, angle between heads and shaft in anterior dorsal ribs: close to 90°(0); low, gentle posteroventral bowing of the base of the shaft (1) (Dilkes & Sues, 2009; Ezcurra, Lecuona & Martinelli, 2010: 51; Dilkes & Arcucci, 2012: 57). This character is inapplicable in taxa that the cervical rib is directed in a high angle to the neck.

The angle between the proximal end and the shaft of the dorsal ribs is approximately 90°in anterior or posterior views in the vast majority of diapsids. However, this angle is considerably lower, resulting in a distinctly transversely broader and dorsoventraly lower trunk than in other archosauromorphs, in Jaxtasuchus salomoni (SMNS 91352A, B) and Doswellia kaltenbachi (USNM 244214). The condition of this character in the other two doswelliids scored here cannot be determined.

368. Dorsal ribs, proximal end of middle dorsal ribs: dichocephalous (0); holocephalous (1) (Laurin, 1991: D3; Reisz, Laurin & Marjanović, 2010: 79; Ezcurra, Lecuona & Martinelli, 2010: 52; Dilkes & Arcucci, 2012: 58; Ezcurra, Scheyer & Butler, 2014: 78; Pritchard et al., 2015: cf. 122).

The proximal end of the middle dorsal ribs possesses a single facet for articulation with its respective vertebra in the vast majority of non-archosauriform diapsids and the archosauriforms Proterosuchus fergusi (GHG 363) and “Chasmatosaurus” yuani (IVPP V2719, V4067). By contrast, the middle dorsal ribs are dichocephalous in Azendohsaurus madagaskarensis (Nesbitt et al., 2015) and non-proterosuchid archosauriforms sampled here.

369. Sacral vertebrae-sacral ribs, ratio between the width of the neural arch + ribs of the first primordial sacral and the length of the neural arch across the zygapophyses: less than three times (0); three times or more (1) (Desojo, Ezcurra & Schultz, 2011: 95).

See comments in Desojo, Ezcurra & Schultz (2011: 860).

370. Sacral vertebrae, number: two (0); three (1); four or more (2) (Reisz & Dilkes, 2003: 48; Reisz, Laurin & Marjanović, 2010: 80; Nesbitt, 2011: 205, 206 and 207; Ezcurra, Scheyer & Butler, 2014: 79), ORDERED (Fig. 29 of Nesbitt, 2011).

See comments in Nesbitt (2011: characters 205, 206 and 207).

371. Sacral ribs: almost entirely restricted to a single sacral vertebra (0); shared between two sacral vertebrae (1) (Nesbitt, 2011: 208) (Fig. 29 of Nesbitt, 2011).

See comments in Nesbitt (2011: character 208).

372. Sacral ribs, anteroposterior length of the first primordial sacral rib versus the second primordial sacral rib in dorsal view: primordial sacral rib one is longer anteroposteriorly than primordial sacral rib two (0); primordial sacral rib two is about the same length or longer anteroposteriorly than primordial sacral rib one (1) (Nesbitt et al., 2015: 216).

See comments in Nesbitt et al. (2015: character 216).

373. Sacral ribs, second rib shape: single unit (0); bifurcates distally into anterior and posterior processes (1) (Dilkes, 1998: 87, in part; Müller, 2004: 105; Ezcurra, Lecuona & Martinelli, 2010: 53; Nesbitt, 2011: 203, in part; Dilkes & Arcucci, 2012: 59; Ezcurra, Scheyer & Butler, 2014: 187; Pritchard et al., 2015: 131) (Fig. 35).

Figure 35 Sacral and caudal vertebrae and ribs of Triassic and Recent saurians in (A, C, E, F) dorsal and (B, D) right lateral views.

(A) Second sacral vertebra of Tanystropheus longobardicus (SMNS 54631); (B) middle caudal vertebrae of Salvator merianae (MACN-He 47992); (C) second sacral vertebra of Mesosuchus browni (SAM-PK-6046); (D) distal caudal vertebrae of Herrerasaurus ischigualastensis (PVSJ 053); (E) second sacral vertebra of Rhynchosaurus articeps (SHYMS 5); and (F) anterior caudal vertebrae of Chanaresuchus bonapartei (PVL 4575). Numbers indicate character-states scored in the data matrix and the arrows indicate anterior direction. Scale bars equal 1 cm in (A, C, E, D), 4 mm in (B), and 2 cm in (F).

See comments in Nesbitt (2011: character 203).

374. Sacral ribs, morphology of posterior process: pointed bluntly (0); pointed sharply (1) (Dilkes, 1998: 87, in part; Pritchard et al., 2015: 132). This character is inapplicable in taxa without a bifurcated second sacral rib (character-state 373-0) (Fig. 35).

See comments in Pritchard et al. (2015: character 132).

375. Sacral and caudal vertebrae, transverse processes and ribs of sacral and/or anterior caudal vertebrae in mature individuals: not fused to each other (0); fused to each other (1) (Rieppel, Mazin & Tchernov, 1999; Müller, 2004: 50; Ezcurra, Scheyer & Butler, 2014: 188) (Fig. 35).

The sacral and caudal vertebrae are not fused to their respective ribs (i.e., the elements can be individualized because of a loose articulation or track of suture) in Petrolacosaurus kansensis (Reisz, 1981), choristoderans (e.g., Cteniogenys sp.: Evans, 1991; Simoedosaurus lemoinei: MNHN.F.BR12153, BR12154), Protorosaurus speneri (BSPG 1995 I 5, cast of WMsN P47361), several erythrosuchids (e.g., Garjainia prima: Huene, 1960; Erythrosuchus africanus: NHMUK PV R3592; Shansisuchus shansisuchus: Young, 1964), and Archeopelta arborensis (CPEZ-239a). In other species scored here, there is no suture visible between the sacral and caudal vertebrae and their respective ribs in mature individuals.

376. Caudal vertebrae, autotomic septa within the centrum: absent (0); present (1) (Gauthier, 1984; Gauthier, Kluge & Rowe, 1988; Pritchard et al., 2015: 135) (Fig. 35).

The presence of autotomous vertebrae is determined by the presence of a partial fissure in the transverse plane approximately at mid-length on the centra of a series of caudal vertebrae. This fissure represents a fracture plane that allows the autotomy of part of the tail and its subsequent regeneration by the growth of a cartilaginous rod. Autotomous caudal vertebrae have been reported in parareptiles, early synapsids (Price, 1940), probably in the early neodiapsid Araeoscelis (Vaughn, 1955), and in lepidosauromorphs (e.g., Gephyrosaurus bridensis: Evans, 1981; Planocephalosaurus robinsonae: Fraser & Walkden, 1984). An autotomic suture has been described on a caudal vertebra of the basal archosauromorph Tanystropheus longobardicus (Wild, 1973), but no evidence for this condition has been subsequently found in articulated, well-preserved specimens (Nosotti, 2007). As a result, the presence of autotomic septa in the caudal vertebrae is restricted to lepidosauromorphs in the taxonomic sample of this analysis.

377. Caudal vertebrae, length of the transverse process + rib versus length across zygapophyses in anterior caudal vertebrae: 0.29–0.41 (0); 0.62–1.20 (1); 1.51–1.68 (2); 2.20–2.72 (3) (Dilkes, 1998: 89; Dilkes & Arcucci, 2012: 56, in part; modified from Ezcurra, Scheyer & Butler, 2014: 191), ORDERED (Fig. 35).

This character describes the proportional transverse length of the transverse process and rib of the anterior caudal vertebrae with respect to the anteroposterior length of the neural arch, measured between the anterior tip of the prezygapophysis and the posterior tip of the postzygapophysis.

378. Caudal vertebrae, distal end of the transverse processes + ribs of anterior caudals in dorsal or ventral view: tapering or squared (0); anteroposteriorly expanded (1) (modified from Dilkes & Arcucci, 2012: 56) (Fig. 35).

The distal end of the anterior caudal ribs or transverse processes and their fused ribs tapers or is squared in dorsal or ventral view in the vast majority of diapsids. By contrast, the distal end of the caudal ribs or transverse processes + ribs are anteroposteriorly expanded, acquiring the process an overall paddle-like contour in dorsal or ventral view, in Tanystropheus longobardicus (PIMUZ T2817), proterochampsids (e.g., Tropidosuchus romeri: PVL 4601; Pseudochampsa ischigualastensis: Trotteyn & Ezcurra, 2014), and Marasuchus lilloensis (PVL 3871).

379. Caudal vertebrae, neural spine height versus anteroposterior length at its base in anterior caudals: 0.66–2.21 (0); 2.36–2.65 (1); 2.92–3.05 (2); 3.42-3.54 (3) (Dilkes, 1998: 88; Müller, 2004: 106; modified from Ezcurra, Scheyer & Butler, 2014: 189), ORDERED.

380. Caudal vertebrae, accessory laminar process on the anterior face of the neural spine on middle caudals: absent (0); present (1) (Benton & Clark, 1988; Nesbitt, 2011: 210) (Fig. 28 of Nesbitt, 2011).

See comments in Nesbitt (2011: character 210).

381. Caudal vertebrae, prezygapophysis of posterior caudals: not elongated (0); elongated more than a quarter of the adjacent centrum (1) (Gauthier, 1986; Nesbitt, 2011: 211) (Fig. 35).

See comments in Nesbitt (2011: character 211).

382. Chevrons, distal anteroposterior width of anterior and middle haemal spines in lateral view: equivalent to proximal width (0); tapering distally (1); longer than proximal width (=paddle-like haemal spine) (2) (Dilkes, 1998: 91; Müller, 2004: 108; modified from Ezcurra, Scheyer & Butler, 2014: 192; Pritchard et al., 2015: 136, in part).

This character describes three different morphologies of the haemal spine of the chevron in lateral view. In most diapsids sampled here, the haemal spine is subrectangular in lateral view (the anteroposterior width of the proximal end is sub-equal to that of the distal end) or tapers distally. In the allokotosaurian Pamelaria dolichotrachela, the haemal spine is anteroposteriorly longer at its distal end than at its proximal end and, as a result, it acquires a paddle-like shape in lateral view (ISI R316).

383. Gastralia: present, forming an extensive ventral basket with closely packed elements (0); present, well separated (1); absent (2) (Dilkes, 1998: 92, in part; Müller, 2004: 109, in part; Nesbitt, 2011: 412; Ezcurra, Scheyer & Butler, 2014: 193, in part; Pritchard et al., 2015: 137, in part; Nesbitt et al., 2015: 238, in part) (Fig. 15).

See comments in Nesbitt (2011: character 412).

384. Scapulocoracoid, both bones fuse with each other in mature individuals: present (0); absent (1) (Rowe, 1989) (Fig. 36).

Figure 36 Scapulae and coracoids of Triassic archosauromorphs in lateral view.

(A) Amotosaurus rotfeldensis (SMNS 50830); (B) Sarmatosuchus otschevi (PIN 2865/68-37); (C) Garjainia prima (PIN 2394/5-32); (D) Euparkeria capensis (SAM-PK-5867, mirrored); (E) Chanaresuchus bonapartei (PVL 4575); and (F) Lewisuchus admixtus (PULR 01, mirrored). Numbers indicate character-states scored in the data matrix and the arrows indicate anterior direction. Scale bars equal 5 mm in (A, D, F), 2 cm in (B, C), and 1 cm in (E).

The scapula and coracoid are fused to each other in Petrolacosaurus kansensis (Reisz, 1981), Youngina capensis (BP/1/3859), lepidosauromorphs (e.g., Gephyrosaurus bridensis: Evans, 1981; Planocephalosaurus robinsonae: Fraser & Walkden, 1984), Protorosaurus speneri (BSPG 1995 I 5, cast of WMsN P47361), Jesairosaurus lehmani (ZAR 09), Trilophosaurus buettneri (Spielmann et al., 2008), Mesosuchus browni (SAM-PK-6536), Prolacerta broomi (BP/1/2675), “Chasmatosaurus” yuani (IVPP V4067), some proterochampsids (Proterochampsa barrionuevoi: Trotteyn, 2011a; Pseudochampsa ischigualastensis: Trotteyn & Ezcurra, 2014), Parasuchus hislopi (ISI R42), and several Triassic avemetatarsalians (e.g., Dimorphodon macronyx: NHMUK PV R41212-13; Marasuchus lilloensis: PVL 3871; Lewisuchus admixtus: PULR 01; Silesaurus opolensis: ZPAL AbIII/2534; Herrerasaurus ischigualastensis: PVSJ 53).

385. Scapulocoracoid, notch on the anterior margin at level of the suture between both bones: absent (0); present (1) (Benton, 2004; Ezcurra, Lecuona & Martinelli, 2010: 57; Nesbitt, 2011: 221) (Fig. 36).

See comments in Nesbitt (2011: character 221).

386. Scapulocoracoid, glenoid fossa orientation: posterolateral (0); posteroventral (1) (Fraser et al., 2002; Nesbitt, 2011: 227) (Figs. 36 and 37).

Figure 37 Scapulae and coracoids of Triassic archosauromorphs in (A) medial, (B) posterolateral, (C) dorsal, and (D) lateral views.

(A) Right scapula and coracoid of Garjainia prima (PIN 2394/5-33); (B) right scapula and coracoid of Gualosuchus reigi (PULR 05); and (C, D) left coracoid of Pamelaria dolichotrachela (ISI R316/1). Numbers indicate character-states scored in the data matrix and the arrows indicate anterior direction. Scale bars equal 2 cm in (A) and 1 cm in (B-D).

See comments in Nesbitt (2011: character 227).

387. Scapula, total length of the scapula versus minimum anteroposterior width of the scapular blade: 1.23–6.73 (0); 7.92–11.31 (1) (Benton, 1985; modified from Ezcurra, Lecuona & Martinelli, 2010: 58; cf. Nesbitt, 2011: 218; cf. Dilkes & Arcucci, 2012: 63) (Figs. 15 and 36).

See comments in Nesbitt (2011: character 218).

388. Scapula, large fenestra between scapula and coracoid immediately anterior to the glenoid region: absent (0); present (1) (New character) (Fig. 36).

The contact between the scapula and coracoid extends along the entire or most of the anterior margin of the bones in the vast majority of diapsids, and only a shallow notch may interrupt them in lateral or medial view. By contrast, the anterior three quarters of the proximal margin of the scapula and the anterior half of the dorsal margin of the coracoid are anteroposteriorly concave in tanystropheids, resulting in a large, oval fenestra between both bones immediately anterior of the glenoid region in lateral or medial view (e.g., Amotosaurus rotfeldensis: SMNS 50830; Macrocnemus bassanii: PIMUZ T4355; Tanystropheus longobardicus: Wild, 1973; Nosotti, 2007).

389. Scapula, strong curvature or inflexion between the proximal end and the posterior margin of the scapular blade: absent (0); present, the angle formed is close to 90°(1) (Benton & Allen, 1997; modified from Pritchard et al., 2015: 145) (Fig. 36).

The scapula of Azendohsaurus madagaskarensis (Nesbitt et al., 2015), Howesia browni (Broom, 1906), Rhynchosaurus articeps (SHYMS 2), erythrosuchids (e.g., Erythrosuchus africanus: NHMUK PV R3592, SAM-PK-905; Garjainia prima: PIN 2394/5), and more crownward archosauriforms (e.g., Euparkeria capensis: SAM-PK-5867; Tropidosuchus romeri: PVL 4604; Batrachotomus kupferzellensis: SMNS 80271) is rather symmetrical in lateral view, with a gradual transition between the glenoid portion and the posterior margin of the base of the scapular blade. By contrast, the posterior margin of the proximal end and the base of the scapular blade form an angle close to 90°in lateral view and, as a result, the scapular blade is strongly asymmetrical, being more posteriorly expanded than anteriorly, in most non-archosauriform diapsids (e.g., Petrolacosaurus kansensis: Reisz, 1981; Youngina capensis: BP/1/3859; Simoedosaurus lemoinei: MNHN.F.BR1009, BR1013; Protorosaurus speneri: BSPG 1995 I 5, cast of WMsN P47361; Amotosaurus rorfeldensis: SMNS 50830; Pamelaria dolichotrachela: ISI R316/1; Mesosuchus browni: SAM-PK-6536; Prolacerta broomi: BP/1/2675; Boreopricea funerea: PIN 3708/1), Proterosuchus alexanderi (NMQR 1484), “Chasmatosaurus” yuani (IVPP V4067), Sarmatosuchus otschevi (PIN 2865/68), and Nundasuchus songeaensis (Nesbitt et al., 2014).

390. Scapula, anterior margin of the scapular blade in lateral view: straight or convex along entire length (0); distinctly concave (1) (DeBraga & Reisz, 1995: 27; Gower & Sennikov, 1997; Reisz, Laurin & Marjanović, 2010: 86; Ezcurra, Lecuona & Martinelli, 2010: 104; Nesbitt, 2011: 217; Ezcurra, Scheyer & Butler, 2014: 85; Nesbitt et al., 2015: 219) (Fig. 36).

See comments in Nesbitt (2011: character 217).

391. Scapula, constriction distal to the glenoid: minimum anteroposterior length greater than half the proximodistal length of the scapula (0); minimum anteroposterior length less than half the proximodistal length of the scapula (1) (modified from Nesbitt et al., 2015: 220) (Figs. 36 and 37).

See comments in Nesbitt et al. (2015: character 220).

392. Scapula, supraglenoid foramen: absent (0); present (1) (Reisz & Dilkes, 2003: 29; Reisz, Laurin & Marjanović, 2010: 87; Ezcurra, Scheyer & Butler, 2014: 86).

The supraglenoid foramen is placed on the posterolateral suface of the scapula immediately above the supraglenoid lip. This foramen is present in Petrolacosaurus kansensis (Reisz, 1981), Gephyrosaurus bridensis (Evans, 1981), and Protorosaurus speneri (Gottmann-Quesada & Sander, 2009) among the species sampled in this analysis.

393. Scapula, lateral tuber on the posterior edge, just dorsal of the glenoid fossa: absent (0); present (1) (Nesbitt, 2011: 219; Pritchard et al., 2015: 146) (Fig. 30 of Nesbitt, 2011).

See comments in Nesbitt (2011: character 219).

394. Scapula, diagonal ridge adjacent to the anterior margin on the medial surface of the scapular blade: absent (0); present (1) (New character) (Fig. 37).

The medial surface of the scapular blade of most diapsids is smooth, but in Petrolacosaurus kansensis (Reisz, 1981), some erythrosuchids (Garjainia prima: PIN 2394/5; Garjainia madiba: BP/1/7152; Erythrosuchus africanus: NHMUK PV R3592), and Smilosuchus gregorii (USNM 18313) there is a diagonal, posterodistally-to-anteroproximally oriented ridge that is adjacent to the anterior margin of the bone and merges with the rest of the medial surface approximately close to the anteroposterior mid-width of the blade. This ridge is placed around mid-length on the scapular blade.

395. Scapula, acromion process: in about the same plane as the ventral edge of the scapula (0); distinctly raised above the ventral edge of the scapula (1) (Nesbitt, 2011: 220) (Figs. 36 and 37).

See comments in Nesbitt (2011: character 220).

396. Scapula, acromion process: gently raised from the anterior margin of the scapular blade (0); sharply raised in an angle between 50–90°from the anterior margin of the scapular blade (1) (modified from Novas, 1992a).

The acromial region of the scapula commonly merges gradually with the scapular neck in the vast majority of diapsids. In Planocephalosaurus robinsonae (Fraser & Walkden, 1984), Riojasuchus tenuisceps (PVL 3827) and Herrerasaurus ischigualastensis (PVSJ 53) the acromion process turns abruptly away from the scapular blade in lateral or medial view, forming an angle between 50°and 90°.

397. Coracoid, anterior border in lateral view: rounded (0); distinctly hooked (1) (Sereno, 1991; Nesbitt, 2011: 226) (Fig. 30 of Nesbitt, 2011).

See comments in Nesbitt (2011: character 226).

398. Coracoid, posterior border in lateral view: unexpanded posteriorly (0); moderately expanded posteriorly (1); strongly expanded posteriorly—the entire border, not only the posteroventral region as is the case in the postglenoid process—and, as a result, the articulated scapula and coracoid are L-shaped in lateral view (2) (New character), ORDERED (Fig. 37).

The entire posterior margin of the coracoid is strongly expanded posteriorly from the level of the glenoid fossa and, as a result, the articulated scapula and coracoid are L-shaped in lateral view in the non-archosauromorph diapsids scored here (e.g., Youngina capensis: BP/1/3859; Planocephalosaurus robinsonae: Fraser & Walkden, 1984; Simoedosaurus lemoinei: MNHN.F.BL9525, BR1018) and Jesairosaurus lehmani (ZAR 09), Trilophosaurus buettneri (Spielmann et al., 2008), and Dimorphodon macronyx (NHMUK PV R41212-13). By contrast, the coracoid of most archosauromorphs is only moderately expanded posteriorly and the expansion is confined to the posteroventral corner of the bone. Only in Pamelaria dolichotrachela (ISI R316/1), Garjainia madiba (cast of NMQR 3051), Yarasuchus deccanensis (ISI R334/49), Tropidosuchus romeri (PVL 4601, 4604), Pseudochampsa ischigualastensis (Trotteyn & Ezcurra, 2014), Parasuchus hislopi (ISI R42), Marasuchus lilloensis (PVL 3871), and Heterodontosaurus tucki (SAM-PK-K1332) the coracoid lacks a posterior expansion.

399. Coracoid, subglenoid lip posterior extension: as developed as or less developed than the supraglenoid lip on the scapula (0); more extended than the supraglenoid lip on the scapula (1) (New character) (Fig. 36).

In the vast majority of diapsids, the supraglenoid lip of the scapula and the subglenoid lip of the coracoid are subequally developed posteriorly or the latter is lower. In Protorosaurus speneri (BSPG 1995 I 5, cast of WMsN P47361), Prolacerta broomi (BP/1/2675), Smilosuchus gregorii (USNM 18313), the ornithosuchids Ornithosuchus longidens (Walker, 1964) and Riojasuchus tenuisceps (PVL 3827), and avemetatarsalians (e.g., Dimorphodon macronyx: NHMUK PV R41212-13; Marasuchus lilloensis: PVL 3871; Heterodontosaurus tucki: SAM-PK-K1332) the subglenoid lip is more posteriorly developed than the supraglenoid lip in lateral or medial view.

400. Coracoid, subglenoid lip lateral extension: poorly developed, resembling the development of the supraglenoid lip on the scapula (0); strongly developed as a shelf-like structure, more developed than the supraglenoid lip on the scapula (1) (New character) (Fig. 37).

The subglenoid lip of the coracoid is strongly laterally developed as a shelf-like structure in the allokotosaurians Pamelaria dolichotrachela (ISI R316/1), Azendohsaurus madagaskarensis (Nesbitt et al., 2015), and Trilophosaurus buettneri (Spielmann et al., 2008). The subglenoid lip possesses only a very limited lateral extension, developed as a mound-like structure in posterior view in other diapsids.

401. Coracoid, biceps process on the lateral surface: absent or small (0); large (1) (Laurin, 1991: H5; Reisz, Laurin & Marjanović, 2010: 88; Nesbitt, 2011: 225; Ezcurra, Scheyer & Butler, 2014: 87) (Figs. 36 and 37).

Nesbitt (2011) scored the presence of a swollen tuber on the posteroventral portion of the coracoid in ornithosuchids, suchians, and ornithodirans (with some absences in taxa deeply nested within those clades, e.g., poposaurids) (see comments in Nesbitt (2011: character 225)). The scorings of this character in the present analysis resemble those of Nesbitt (2011), but it is found here that a large, swollen biceps tubercle is also present in proterochampsids with well-preserved coracoids (Gualosuchus reigi: PULR 05, PVL 4576; Chanaresuchus bonapartei: MCZ 4035).

402. Coracoid, postglenoid process separated from the glenoid fossa by a notch: absent (0); present (1) (Clark et al., 2004; modified from Nesbitt, 2011: 222; modified from Dilkes & Arcucci, 2012: 64) (Fig. 36).

See comments in Nesbitt (2011: character 222).

403. Coracoid, postglenoid process shape in lateral view: rounded posterior margin (0); tapering posterior margin (1) (New character). This character is inapplicable in taxa that lack a postglenoid process.

The postglenoid process of the coracoid usually tapers posteriorly in archosauromorphs, but it possesses a more rounded margin in the enigmatic archosauriform Youngosuchus sinensis (IVPP V3239) and the archosaurs Nundasuchus songeaensis (Nesbitt et al., 2014), Batrachotomus kupferzellensis (SMNS 80271), Prestosuchus chiniquensis (UFRGS-PV-0629-T), and Marasuchus lilloensis (PVL 3871).

404. Cleithrum: present (0); absent (1) (Laurin, 1991: E11; Reisz, Laurin & Marjanović, 2010: 85; Ezcurra, Scheyer & Butler, 2014: 84; Pritchard et al., 2015: 140).

The cleithrum is present in the early diapsids Petrolacosaurus kansensis (Reisz, 1981) and Acerosodontosaurus piveteaui (Bickelmann, Müller & Reisz, 2009), but it is absent in choristoderans and all sampled saurians. The presence of this bone in Youngina capensis is uncertain (Smith & Evans, 1996).

405. Interclavicle: present (0); absent (1) (Gauthier, 1986; Benton, 2004; Ezcurra, Lecuona & Martinelli, 2010: 54, in part; Nesbitt, 2011: 214) (Fig. 15).

See comments in Nesbitt (2011: character 214).

406. Interclavicle, anterior process: present (0); absent (1) (Laurin, 1991: J6; DeBraga & Reisz, 1995: 26; Müller, 2004: 55; Reisz, Laurin & Marjanović, 2010: 82; Ezcurra, Lecuona & Martinelli, 2010: 54, in part; Dilkes & Arcucci, 2012: 61, in part; Ezcurra, Scheyer & Butler, 2014: 81) (Fig. 38).

Figure 38 Interclavicles of Triassic archosauromorphs in ventral view.

(A) Prolacerta broomi (BP/1/2675); (B) Tasmaniosaurus triassicus (UTGD 54655); (C) Proterosuchus fergusi (GHG 363); (D) Garjainia prima (PIN 2394/5-34); and (E) Parasuchus hislopi (ISI R42). Numbers indicate character-states scored in the data matrix and the arrows indicate anterior direction. Scale bars equal 1 cm.

This character describes the presence of a median, anterior process on the anterior margin of the interclavicle, which gives to the bone the shape of a cross in dorsal or ventral view. The anterior process of the interclavicle is present in disparate diapsid groups and may be phylogenetically informative at low taxonomic levels.

407. Interclavicle, anterior margin with a median notch: absent (0); present (1) (modified from Dilkes, 1998: 97; Müller, 2004: 111; Ezcurra, Lecuona & Martinelli, 2010: 55; Dilkes & Arcucci, 2012: 61, in part; Ezcurra, Scheyer & Butler, 2014: 195; Pritchard et al., 2015: 143) (Fig. 38).

A median, well-defined notch on the anterior margin of the interclavicle is present in Petrolacosaurus kansensis (Reisz, 1981), Simoedosaurus lemoinei (MNHN.F.R1413) and several early archosauromorphs, such as Macrocnemus bassanii (PIMUZ T4355), Jesairosaurus lehmani (ZAR 06), Prolacerta broomi (BP/1/2675), Proterosuchus fergusi (GHG 363), Proterosuchus alexanderi (NMQR 1484), and Tasmaniosaurus triassicus (UTGD 54655). By contrast, the anterior margin of the interclavicle is transversely convex or gently concave, straight, or tapers anteriorly in ventral or dorsal view in Youngina capensis (BP/1/3859), Protorosaurus speneri (Gottmann-Quesada & Sander, 2009), allokotosaurians (e.g., Azendohsaurus madagaskarensis: Nesbitt et al., 2015; Trilophosaurus buettneri: Spielmann et al., 2008), rhynchosaurs (e.g., Mesosuchus browni: SAM-PK-6536; Rhynchosaurus articeps: NHMUK PV R1239),“Chasmatosaurus” yuani (IVPP V4067), and more crownward archosauriforms (e.g., Doswellia kaltenbachi: USNM 244214; Parasuchus hislopi: ISI R42; Aetosauroides scagliai: PVL 2073).

408. Interclavicle, lateral processes: well developed (0); reduced or absent (1) (Gauthier, 1984; modified from Nesbitt, 2011: 215; cf. Dilkes & Arcucci, 2012: 60) (Fig. 38).

See comments in Nesbitt (2011: character 215).

409. Interclavicle, webbed between lateral and posterior processes: present, proximal half of the bone subtriangular or diamond-shaped (0); absent, rather sharp angles between processes (1) (Laurin, 1991: J6; Reisz, Laurin & Marjanović, 2010: 83; Ezcurra, Scheyer & Butler, 2014: 82; Pritchard et al., 2015: cf. 142) (Fig. 38).

This character accounts for the shape of the anterior end of the interclavicle in dorsal or ventral view. In Youngina capensis (BP/1/3859, SAM-PK-K7710), lepidosauromorphs (e.g., Gephyrosaurus bridensis: Evans, 1981), Simoedosaurus lemoinei (MNHN.F.R1413), Azendohsaurus madagaskarensis (Nesbitt et al., 2015), Howesia browni (Broom, 1906: plate 15), and proterosuchids (Proterosuchus fergusi: GHG 363; Proterosuchus alexanderi: NMQR 1484; “Chasmatosaurus” yuani: IVPP V4067) the lateral processes of the interclavicle turn abruptly away from the posterior ramus of the bone, forming angles of approximately 90°between them. By contrast, the anterior end of the interclavicle is subtriangular (with a posterior apex) or diamond-shaped because of a gradual transition between the posterior ramus and the lateral processes of the bone.

410. Interclavicle, transverse width at mid-length of the posterior process versus the length of the posterior process: 0.07–0.14 (0); 0.20–0.27 (1) (Laurin, 1991: J6; Reisz, Laurin & Marjanović, 2010: 84; modified from Ezcurra, Scheyer & Butler, 2014: 83), ORDERED (Fig. 38).

The posterior ramus of the interclavicle is a flat and strap-like bone, in which the ratio of its transverse width to its anteroposterior length being less than 0.15, in non-archosauriform diapsids and some archosauriforms (e.g., Euparkeria capensis: SAM-PK-5867; Doswellia kaltenbachi: USNM 244214; Aetosauroides scagliai: PVL 2073). By contrast, this ramus is considerably broader and developed as a plate-like process in proterosuchids (Proterosuchus fergusi: GHG 363; Proterosuchus alexanderi: NMQR 1484; “Chasmatosaurus” yuani: IVPP V4067), erythrosuchids (Garjainia prima: PIN 2394/5), phytosaurs (e.g., Parasuchus hislopi: ISI R42), and Prestosuchus chiniquensis (UFRGS-PV-0152-T).

411. Interclavicle, posterior ramus: little change in width along entire length (0); gradual transverse expansion present (1) (Dilkes, 1998: 98; Müller, 2004: 112; Ezcurra, Lecuona & Martinelli, 2010: 56; Dilkes & Arcucci, 2012: 62; Ezcurra, Scheyer & Butler, 2014: 196; Pritchard et al., 2015: 144) (Fig. 38).

The posterior ramus of the interclavicle possesses a transverse, gradual expansion around the posterior third of the process and the lateral margins converge medially at the posterior end in Simoedosaurus lemoinei (MNHN.F.R1413), Jesairosaurus lehmani (ZAR 06), allokotosaurians (Azendohsaurus madagaskarensis: Nesbitt et al., 2015; Trilophosaurus buettneri: Spielmann et al., 2008), rhynchosaurs (e.g., Mesosuchus browni: SAM-PK-6536; Rhynchosaurus articeps: SHYMS 2), Prolacerta broomi (BP/1/2675), Tasmaniosaurus triassicus (UTGD 54655), Garjainia prima (PIN 2394/5), Doswellia kaltenbachi (USNM 244214), phytosaurs (e.g., Parasuchus hislopi: ISI R42; Smilosuchus gregorii: USNM 18313), Ornithosuchus longidens (Walker, 1964) and Prestosuchus chiniquensis (UFRGS-PV-0152-T). In other diapsids that possess an interclavicle and sampled here, the lateral margins of the posterior ramus are subparallel to each other along its entire length.

412. Clavicle, articulation with interclavicle: on the anteroventral surface of the interclavicle (0); on the anterodorsal surface of the interclavicle (1); into a deep, anteriorly facing socket (2) (Rieppel, Mazin & Tchernov, 1999; Müller, 2004: 53; reworded and modified from from Ezcurra, Scheyer & Butler, 2014: 194) (Fig. 15).

In the vast majority of diapsids, the clavicle articulates on the anteroventral surface of the lateral process of the interclavicle. But this articulation is reversed, with the clavicle articulating on the anterodorsal surface of the lateral process of the interclavicle, in Trilophosaurus buettneri (Gregory, 1945) and Nundasuchus songeaensis (Nesbitt et al., 2014). A particular condition occurs in Doswellia kaltenbachi (USNM 244214) and Parasuchus hislopi (ISI R42), in which the clavicle fits into a deep, anteriorly facing socket in the interclavicle.

413. Sternum: not mineralized (0); mineralized (bone or calcified cartilage) (1) (Laurin, 1991: A5; Müller, 2004: 81; Reisz, Laurin & Marjanović, 2010: 81; Ezcurra, Scheyer & Butler, 2014: 80; Pritchard et al., 2015: 149). This character is scored as missing data in taxa without well preserved, articulated specimens.

The presence of a mineralized sternum has been described in Youngina capensis (Smith & Evans, 1996), Jesairosaurus lehmani (Jalil, 1997), and Boreopricea funerea (Tatarinov, 1978) among the species sampled here. In addition, a mineralized sternum is also reported here in the tanystropheid Amotosaurus rotfeldensis (SMNS 54810) and is represented by a pair of sternal plates with an extensive median suture between each other. In other diapsids sampled here, the absence of a mineralized sternum is difficult to determine if there is no articulated specimen and, as a result, the presence of sternal plates may be more widely distributed than currently documented.

414. Forelimb-hindlimb, length ratio: >0.55 (0); <0.55 (1) (modified from Gauthier, 1984; Nesbitt, 2011: 212) (Fig. 15).

See comments in Nesbitt (2011: character 212).

415. Humerus, torsion between proximal and distal ends: approximately 45°or more (0); 35°or less (1) (Müller, 2004: 145; modified from Ezcurra, Scheyer & Butler, 2014: 197) (Fig. 39).

Figure 39 Humeri of Permo-Triassic neodiapsids in (A–D, F) anterior/dorsal and (E) posterior/ventral views.

(A) Right element of Youngina capensis (BP/1/3859); (B) right element of Pamelaria dolichotrachela (ISI R322); (C) right element of “Chasmatosaurus” yuani (IVPP V2719); (D) right element of Erythrosuchus africanus (SAM-PK-905); (E) left element of Parasuchus hislopi (ISI R42, mirrored); and (F) left element of Silesaurus opolensis (ZPAL AbIII/452, mirrored). Numbers indicate character-states scored in the data matrix. Scale bars equal 2 mm in (A), 1 cm in (B, C, E, F), and 2 cm in (D).

The main axes of the proximal and distal ends of the humerus form an angle of approximately 45°or more (usually closer to 90°) between each other in the non-archosauromorph diapsids Petrolacosaurus kansensis (Reisz, 1981), Youngina capensis (BP/1/3859), Planocephalosaurus robinsonae (Fraser & Walkden, 1984), Gephyrosaurus bridensis (Evans, 1981) and Simoedosaurus lemoinei (MNHN.F.BL9626, BR1381), and the archosauromorphs Prolacerta broomi (BP/1/2675), Boreopricea funerea (PIN 3708/1), “Chasmatosaurus” yuani (IVPP V2719, V4067) and Vancleavea campi (AMNH 30884 cast). In other archosauromorphs sampled here, the main axes of the proximal and distal ends of the humerus extend subparallel to each other.

416. Humerus, transverse width of the proximal end versus total length of the bone in mature individuals: 0.20–0.41 (0); 0.44–0.70 (1) (modified from Nesbitt, 2007; Nesbitt, 2011: 236) (Fig. 39).

See comments in Nesbitt (2011: character 236).

417. Humerus, proximal articular surface in proximal view: subrectangular to crescent-shape (0); sub-oval (1) (modified from Bonaparte, 1991).

The proximal articular surface of the humerus is subrectangular (with rather straight anterior and posterior margins) or crescent-shape (with a transversely concave anterior margin) in proximal view in the vast majority of diapsids. By contrast, the anterior margin of the proximal articular surface of the humerus is straight or gently transversely convex and the posterior one is strongly convex in proximal view in Simoedosaurus lemoinei (MNHN.F.BL9626, BR1381), Archeopelta arborensis (CPEZ-239a), Parasuchus hislopi (ISI R42), and Marasuchus lilloensis (PVL 3871). The proximal articular surface of the humerus in these taxa acquires a suboval outline in proximal view.

418. Humerus, proximal articular surface: continuous with the deltopectoral crest (0); separated by a gap from the deltopectoral crest (1) (Nesbitt, 2011: 233) (Fig. 31 of Nesbitt, 2011).

See comments in Nesbitt (2011: character 233).

419. Humerus, proximal end in anterior view: approximately symmetric (0); medially expanded, being asymmetric (1) (New character) (Fig. 39).

The proximal end of the humerus is rather symmetric in anterior or posterior view, being similarly medially and laterally expanded, in most diapsids. In some proterochampsids (e.g., Gualosuchus reigi: PVL 4576; Chanaresuchus bonapartei: MCZ 4035, PVL 4575), doswelliids (Archeopelta arborensis: CPEZ-239a; Jaxtasuchus salomoni: SMNS 91002), phytosaurs (e.g., Parasuchus hislopi: ISI R42; Smilosuchus gregorii: USNM 18313), ornithosuchids (Ornithosuchus longidens: Walker, 1964; Riojasuchus tenuisceps: PVL 3827), Nundasuchus songeaensis (Nesbitt et al., 2014), and avemetatarsalians (e.g., Dimorphodon macronyx: NHMUK PV R41212-13; Marasuchus lilloensis: PVL 3871; Silesaurus opolensis: ZPAL AbIII/452; Heterodontosaurus tucki: SAM-PK-K1332; Herrerasaurus ischigualastensis: MACN-Pv 18060, PVSJ 373), the proximal end of the humerus is considerably more expanded medially than laterally and, as a result, this portion of the bone is asymmetrical in anterior or posterior view.

420. Humerus, conical process on the proximal surface, placed immediately adjacent to the base of the deltopectoral crest: absent (0); present (1) (New character) (Fig. 39).

A large, conical process is present on the lateral end of the proximal surface of the humerus and immediately adjacent to the proximal end of the deltopectoral crest in Pamelaria dolichotrachela (ISI R316/1, R322), Azendohsaurus madagaskarensis (Nesbitt et al., 2015) and Prolacerta broomi (BP/1/2675). In other diapsids sampled here this process is absent.

421. Humerus, internal tuberosity distinctly separated from the proximal articular surface in anterior or posterior views: absent (0); present (1) (New character) (Fig. 39).

The internal tuberosity of the humerus is separated from the proximal articular surface by a non-articular and concave surface or a distinct change in slope between both regions in anterior view in Simoedosaurus lemoinei (MNHN.F.BR1381), Trilophosaurus buettneri (USNM mounted partial skeleton), doswelliids (Archeopelta arborensis: CPEZ-239a; Jaxtasuchus salomoni: SMNS 91002), some phytosaurs (Parasuchus hislopi: ISI R42; Smilosuchus gregorii: USNM 18313), ornithosuchids (Ornithosuchus longidens: Walker, 1964; Riojasuchus tenuisceps: PVL 3827), Nundasuchus songeaensis (Nesbitt et al., 2014), Aetosauroides scagliai (PVL 2073), loricatans (e.g., Prestosuchus chiniquensis: UFRGS-PV-0152-T; Batrachotomus kupferzellensis: SMNS 80276), and several early avemetatarsalians (e.g., Dimorphodon macronyx: NHMUK PV R41212-13; Marasuchus lilloensis: PVL 3871; Herrerasaurus ischigualastensis: MACN-Pv 18060, PVSJ 373). By contrast, the internal tuberosity merges gradually, without a distinct differentiation, with the proximal articular surface of the bone in other diapsids.

422. Humerus, shape of the deltopectoral crest in lateral view: rounded or subtriangular (0); subrectangular or trapezoidal (1) (modified from Sereno, 1991; Juul, 1994; Ezcurra, Lecuona & Martinelli, 2010: 118).

The deltopectoral crest is rounded (with convex proximal and distal ends that merge gradually with the rest of the bone in lateral view) or subtriangular (with a distinct apex close to its mid-length) in most diapsids. By contrast, the deltopectoral crest possesses two distinct apexes, resulting in a subrectangular or trapezoidal contour in lateral view, in several proterosuchians (e.g., “Chasmatosaurus” yuani: IVPP V2719, V4067; Garjainia prima: PIN 951/36; Garjainia madiba: BP/1/7336; Erythrosuchus africanus: SAM-PK-905), Nicrosaurus kapffi (Huene, 1923), Gracilisuchus stipanicicorum (CRILAR-Pv 490), and Herrerasaurus ischigualastensis (MACN-Pv 18060, PVSJ 373).

423. Humerus, ventral margin of the deltopectoral crest developed as a thick subcilindrical tuberosity that is well differentiated from the thinner dorsal margin: present (0); absent (1) (New character) (Fig. 39).

The ventral margin of the deltopectoral crest is thickened in comparison with the rest of the structure and this region is developed as a subcilindrical tuberosity in most non-archosauriform archosauromorphs (with the exception of tanystropheids, e.g., Tanystropheus longobardicus: PIMUZ T2817) and proterosuchians (e.g., Proterosuchus alexanderi: NMQR 1484; “Chasmatosaurus” yuani: IVPP V2719, V4067; Garjainia prima: PIN 951/36; Garjainia madiba: BP/1/7336; Erythrosuchus africanus: SAM-PK-905). In other archosauriforms, the ventral margin of the deltopectoral crest possesses a rather sharp edge that is similar in depth to the rest of the crest.

424. Humerus, length of the deltopectoral crest relative to total length of the bone in mature individuals: 0.16–0.18 (0); 0.24–0.49 (1); 0.52–0.55 (2) (modified from Benton, 1990; Juul, 1994; modified from Ezcurra, Lecuona & Martinelli, 2010: 119; modified from Nesbitt, 2011: 230), ORDERED (Fig. 39).

This character accounts for the total length of the deltopectoral crest relative to the total length of the humerus. By contrast, Nesbitt (2011: character 230) considered the length of the deltopectoral crest from its proximal end up to its apex. As a result, the scorings of this character and those of character 230 of Nesbitt (2011) strongly differ from each other, and both characters are probably independent from each other.

425. Humerus, entepicondyle size in mature individuals: moderately large (0); strongly developed (1) (Laurin, 1991: I2; Reisz, Laurin & Marjanović, 2010: 90; Ezcurra, Scheyer & Butler, 2014: 89) (Fig. 39).

The entepicondyle of the humerus is strongly medially developed, being transversely broader than the minimum width of the shaft , in most non-archosauriform diapsids (e.g., Youngina capensis: BP/1/3859; Acerosodontosaurus piveteaui: MNHN 1908-32-57; Planocephalosaurus robinsonae: Fraser & Walkden, 1984; Simoedosaurus lemoinei: MNHN.F.BL9626, BR1381; Aenigmastropheus parringtoni: UMZC T836; Protorosaurus speneri: BSPG 1995 I 5, cast of WMsN P47361; Pamelaria dolichotrachela: ISI R316/1, R322; Trilophosaurus buettneri: Spielmann et al., 2008; Mesosuchus browni: SAM-PK-6536; Rhynchosaurus articeps: SHYMS 2), “Chasmatosaurus” yuani (IVPP V2719, V4067), Garjainia prima (PIN 951/36), Garjainia madiba (BP/1/7336), Erythrosuchus africanus (SAM-PK-905), phytosaurs (e.g., Parasuchus hislopi: ISI R42; Smilosuchus gregorii: USNM 18313; Nicrosaurus kapffi: Huene, 1923), and Aetosauroides scagliai (PVL 2073, 2091).

426. Humerus, entepicondylar foramen: present (0); absent (1) (Laurin, 1991: F6; Reisz, Laurin & Marjanović, 2010: 91; Ezcurra, Scheyer & Butler, 2014: 90; Pritchard et al., 2015: 153) (Fig. 39).

An entepicondylar foramen that completely pierces the distal end of the humerus is present in Petrolacosaurus kansensis (Reisz, 1981), Youngina capensis (BP/1/3859), Acerosodontosaurus piveteaui (MNHN 1908-32-57), lepidosauromorphs (Planocephalosaurus robinsonae: Fraser & Walkden, 1984; Gephyrosaurus bridensis: Evans, 1981), and “Chasmatosaurus” yuani (IVPP V2719, V4067). The entepicondylar foramen is not completely enclosed in the proterosuchid species.

427. Humerus, ectepicondylar region: foramen present (0); foramen absent, supinator process and groove present (1); supinator process, groove or foramen absent (2) (Laurin, 1991: J7; Müller, 2004: 64, 176; Reisz, Laurin & Marjanović, 2010: 92; Nesbitt, 2011: 234, in part; Dilkes & Arcucci, 2012: 65, in part; modified from Ezcurra, Scheyer & Butler, 2014: 91; Pritchard et al., 2015: 150, 151) (Fig. 39).

See comments in Nesbitt (2011: character 234).

428. Humerus, capitellum (radial condyle) and trochlea (ulnar condyle): ball-shaped structures distinct from the ectepicondyle and entepicondyle (0); absent or incipient (1) (Rieppel, Mazin & Tchernov, 1999; Müller, 2004: 63; modified from Ezcurra, Scheyer & Butler, 2014: 198; Pritchard et al., 2015: 156) (Fig. 39).

The radial and ulnar condyles of the distal end of the humerus are developed as strongly convex, ball-shaped articular surfaces in Petrolacosaurus kansensis (Reisz, 1981), Youngina capensis (BP/1/3859), Acerosodontosaurus piveteaui (MNHN 1908-32-57), Simoedosaurus lemoinei (MNHN.F.BL9626, BR1381), Aenigmastropheus parringtoni (UMZC T836), Protorosaurus speneri (BSPG 1995 I 5 cast of WMsN P47361, BSPG AS VII 1207), allokotosaurians (Pamelaria dolichotrachela: ISI R316/1, R322; Trilophosaurus buettneri: (Spielmann et al., 2008); Azendohsaurus madagaskarensis: Nesbitt et al., 2015), and Dimorphodon macronyx (Padian, 1983). In the vast majority of archosauromorphs, the articular facets of the distal end of the humerus are not differentiated into discrete, convex condyles or are incipiently developed.

429. Humerus, trochlea (ulnar condyle) situated approximately at mid-width on the distal end of the bone: present (0); absent, being considerably laterally displaced (1) (Ezcurra, Scheyer & Butler, 2014: 199). This character is inapplicable for taxa that lack or have incipient radial and ulnar condyles (Fig. 39).

Among the taxa that possess a well-developed, ball-like ulnar condyle, this articular structure is placed approximately at mid-width on the distal end of the humerus in Youngina capensis (BP/1/3859), Acerosodontosaurus piveteaui (MNHN 1908-32-57), Simoedosaurus lemoinei (MNHN.F.BL9626, BR1381), and Aenigmastropheus parringtoni (UMZC T836). In other speices, the ulnar condyle is considerably laterally displaced from the point of mid-width of the distal end of the bone (e.g., Petrolacosaurus kansensis: Reisz, 1981; Protorosaurus speneri: BSPG 1995 I 5 cast of WMsN P47361, BSPG AS VII 1207; Pamelaria dolichotrachela: ISI R316/1, R322; Trilophosaurus buettneri: Spielmann et al., 2008; Azendohsaurus madagaskarensis: Nesbitt et al., 2015).

430. Ulna, olecranon process: absent, not ossified or very low (0); prominent but lower than its anteroposterior depth at base (1); strongly developed, being higher than its anteroposterior depth at base (2) (Laurin, 1991: C2; DeBraga & Rieppel, 1997; Müller, 2004: 147; modified from Reisz, Laurin & Marjanović, 2010: 95; Ezcurra, Scheyer & Butler, 2014: 94; Pritchard et al., 2015: 157, in part), ORDERED (Fig. 40).

Figure 40 Anterior zeugopodia and autopodia of Triassic and Early Jurassic archosauromorphs in (A, D) anterior/dorsal and (B, C) posterior/ventral views.

(A) Left distal halves of the ulna and radius, carpal region and partial manus of Riojasuchus tenuisceps (PVL 3827); (B) right radius of Erythrosuchus africanus (SAM-PK-905); (C) right ulna and radius of Macrocnemus bassanii (PIMUZ T4355); and (D) right lower forelimb of Heterodontosaurus tucki (SAM-PK-K1332). Close up of the carpals in anterior/dorsal view in (D). Numbers indicate character-states scored in the data matrix. Scale bars equal 5 mm in (A, C, D) and 2 cm in (B).

The olecranon process of the ulna is strongly developed, being higher than its anteroposterior depth at base, in Petrolacosaurus kansensis (Reisz, 1981), some specimens of Simoedosaurus lemoinei (MNHN.F.BR12108), Aenigmastropheus parringtoni (UMZC T836), and Protorosaurus speneri (BSPG 1995 I 5, cast of WMsN P47361). In Cteniogenys sp. (Evans, 1991), some specimens of Simoedosaurus lemoinei (SMNS 58577), Amotosaurus rotfeldensis (SMNS 54783), Azendohsaurus madagaskarensis (Nesbitt et al., 2015), Trilophosaurus buettneri (USNM mounted skeleton), one specimen of “Chasmatosaurus” yuani (IVPP V4067), Vancleavea campi (Nesbitt et al., 2009), Smilosuchus gregorii (USNM 18313), Riojasuchus tenuisceps (PVL 3827), suchians (e.g., Turfanosuchus dabanensis: IVPP V3237; Aetosauroides scagliai: PVL 2073, 2091; Batrachotomus kupferzellensis: SMNS 80275) and several basal avemetatarsalians (e.g., Marasuchus lilloensis: PVL 3871; Heterodontosaurus tucki: SAM-PK-K1332; Herrerasaurus ischigualastensis: PVSJ 373), the olecranon process is prominent, but lower than the depth at its base. By contrast, the olecranon process is not ossified or very low in the other diapsids sampled here.

431. Ulna, olecranon process in lateral view: tapering towards the proximal tip of the bone (0); subrectangular or slightly expanded towards the proximal tip of the bone (1) (Ezcurra, Scheyer & Butler, 2014: 201). This character is inapplicable in taxa without an olecranon process (e.g., Macrocnemus).

The olecranon process of most diapsids continuously tapers towards its distal end in lateral view. However, the olecranon process is subrectangular, with sub-parallel lateral and medial margins in lateral view, or expands slightly anteroposteriorly towards the proximal end of the bone in Simoedosaurus lemoinei (MNHN.F.BR12108), proterosuchians (e.g., Proterosuchus fergusi: SAM-PK-K140; Proterosuchus alexanderi: NMQR 1484; “Chasmatosaurus” yuani: IVPP V4067; Sarmatosuchus otschevi: PIN 2865/68; Cuyosuchus huenei: MCNAM PV 2669; Erythrosuchus africanus: SAM-PK-905), and the loricatans Prestosuchus chiniquensis (UFRGS-PV-0152-T) and Batrachotomus kupferzellensis (SMNS 80275).

432. Ulna, olecranon process as a separate ossification: absent (0); present (1) (Laurin, 1991: C2; Ezcurra, Scheyer & Butler, 2014: 200).

The olecranon process represents a separate ossification from the rest of the ulna in Protorosaurus speneri (Gottmann-Quesada & Sander, 2009) and some specimens of Amotosaurus rotfeldensis (SMNS 54783, 54810). In all other diapsids scored here, the olecranon process and the rest of the ulna are co-ossified.

433. Ulna, lateral tuber (=radius tuber) on the proximal portion: absent (0); present (1) (Nesbitt, 2011: 237; Dilkes & Arcucci, 2012: 67) (Figs. 40 and 31 of Nesbitt, 2011).

See comments in Nesbitt (2011: character 237).

434. Ulna, distal end in posterolateral view: rounded and convex (0); squared off where the distal surface is nearly flat (1) (Nesbitt, 2011: 238; Dilkes & Arcucci, 2012: 66) (Fig. 31 of Nesbitt, 2011).

See comments in Nesbitt (2011: character 238).

435. Radius, total length versus total length of the humerus: 0.62–0.66 (0); 0.69–0.92 (1); 0.95–0.97 (2); 1.12–1.17 (3) (Laurin, 1991: A6, H7; Reisz, Laurin & Marjanović, 2010: 93; modified from Nesbitt, 2011: 241; modified from Ezcurra, Scheyer & Butler, 2014: 92), ORDERED (Fig. 15).

See comments in Nesbitt (2011: character 241).

436. Radius, length in comparisons with that of the ulna: shorter (0); longer or subequal (1) (Rieppel, Mazin & Tchernov, 1999; Müller, 2004: 66; Ezcurra, Scheyer & Butler, 2014: 202). The olecranon process of the ulna should not be taken into account for this character if present.

437. Radius, shape: straight (0); twisted in lateral view (1) (Laurin, 1991: I3; Reisz, Laurin & Marjanović, 2010: 94; Ezcurra, Scheyer & Butler, 2014: 93).

The radius is a straight or slightly bowed bone in the vast majority of diapsids, but it is twisted (or sigmoid) in lateral view in Youngina capensis (Gow, 1975), Acerosodontosaurus piveteaui (Bickelmann, Müller & Reisz, 2009), and Simoedosaurus lemoinei (Sigogneau-Russell, 1981).

438. Radius, distal end: unexpanded or poorly anteroposteriorly expanded (0); strongly anteroposteriorly expanded (1) (New character) (Fig. 40).

The radius of erythrosuchids (Guchengosuchus shiguaiensis: Peng, 1991; Garjainia prima: Huene, 1960; Erythrosuchus africanus: SAM-PK-905; Shansisuchus shansisuchus: Young, 1964) and Riojasuchus tenuisceps (PVL 3828) is strongly expanded anteroposteriorly at its distal end, being considerably deeper than the proximal end. In other diapsids, the radius is not expanded or only slightly expanded anteroposteriorly.

439. Carpals, intermedium: present (0); absent (1) (Gauthier, 1984; Pritchard et al., 2015: 160) (Fig. 40).

The intermedium is present in most diapsids scored here, but it is has been described as absent in the dinosaurs Herrerasaurus ischigualastensis (Sereno, 1994; PVSJ 373) and Heterodontosaurus tucki (Santa Luca, 1980; SAM-PK-K1332).

440. Carpals, perforating foramen between intermedium and ulnare: present (0); absent (1) (Benton, 1985; Pritchard et al., 2015: 161). This character is inapplicable in taxa that lack an intermedium.

A fully enclosed perforating foramen between the intermedium and ulnare is present in all the non-archosauriform diapsids sampled here, such as Petrolacosaurus kansensis (Reisz, 1981), Acerosodontosaurus piveteaui (MNHN 1908-32-57), Tanystropheus longobardicus (Nosotti, 2007), Azendohsaurus madagaskarensis (Nesbitt et al., 2015), and Noteosuchus colletti (Carroll, 1976). By contrast, the contact between the intermedium and ulnare is not interrupted by a perforating foramen in the three archosauriforms in which this character could be scored, namely Proterosuchus fergusi (SAM-PK-K140), the erythrosuchid GHG 7433MI, and Riojasuchus tenuisceps (PVL 3827).

441. Carpals, medial centrale: present (0); absent (1) (Laurin, 1991: F7; Reisz, Laurin & Marjanović, 2010: 96; Ezcurra, Scheyer & Butler, 2014: 95; Pritchard et al., 2015: 158).

The medial centrale is present in all the non-archosaurian diapsids scored here, with the exception of Tanystropheus longobardicus (Nosotti, 2007). This carpal is absent in the dinosaurs Herrerasaurus ischigualastensis (Sereno, 1994; PVSJ 373) and Heterodontosaurus tucki (Santa Luca, 1980; SAM-PK-K1332), and the condition could not be determined in any of the sampled proterochampsids, doswelliids, phytosaurs, ornithosuchids, and suchians.

442. Carpals, lateral centrale: large (0); small or absent (1) (Laurin, 1991: E12; Reisz, Laurin & Marjanović, 2010: 97; Ezcurra, Scheyer & Butler, 2014: 96) (Fig. 40).

The vast majority of the sampled diapsids possess a large lateral centrale, but this carpal bone is small or absent in Tanystropheus longobardicus (Nosotti, 2007) and Herrerasaurus ischigualastensis (Sereno, 1994; PVSJ 373).

443. Carpals, pisiform: present (0); absent (1) (New character) (Fig. 40).

The pisiform is interpreted as absent in the carpus of Youngina capensis (Gow, 1975), Protorosaurus speneri (BSPG 1995 I 5, cast of WMsN P47361), Tanystropheus longobardicus (Nosotti, 2007), Azendohsaurus madagaskarensis (Nesbitt et al., 2015), Noteosuchus colletti (Carroll, 1976), and Herrerasaurus ischigualastensis (Sereno, 1994; PVSJ 373). Conversely, this bone is present in Petrolacosaurus kansensis (Reisz, 1981), Acerosodontosaurus piveteaui (Bickelmann, Müller & Reisz, 2009), Trilophosaurus buettneri (Nesbitt et al., 2015), and Heterodontosaurus tucki (Santa Luca, 1980).

444. Carpals, distal carpal five: absent (0); present (1) (Merck, 1997; Pritchard et al., 2015: 159) (Fig. 40).

The fifth distal carpal is present in the non-archosauromorph diapsids Petrolacosaurus kansensis (Reisz, 1981), Acerosodontosaurus piveteaui (Bickelmann, Müller & Reisz, 2009) and Youngina capensis (Gow, 1975), and the dinosaurs Heterodontosaurus tucki (Santa Luca, 1980) and Herrerasaurus ischigualastensis (Ezcurra, 2010a). By contrast, this bone is absent in the non-archosauriform archosauromorphs Protorosaurus speneri (Gottmann-Quesada & Sander, 2009), Tanystropheus longobardicus (Nosotti, 2007), Azendohsaurus madagaskarensis (Nesbitt et al., 2015), Trilophosaurus buettneri (Nesbitt et al., 2015), and Mesosuchus browni (Dilkes, 1998).

445. Manus, longest metacarpal + digit: longer than humeral length (0); subequal to shorter than humeral length (1) (Senter, 2004: 51; Ezcurra, Scheyer & Butler, 2014: 203).

The sum of the lengths of the longest metacarpal and digit is greater than that of the humerus in the early neodiapsids Petrolacosaurus kansensis (Reisz, 1981) and Youngina capensis (Smith & Evans, 1996; SAM-PK-K7710) and in the dinosaur Herrerasaurus ischigualastensis (Sereno, 1994; PVSJ 373).

446. Metacarpus, length of the longest metacarpal versus length of the longest metatarsal: 0.34–0.39 (0); 0.43–0.45 (1); 0.54–0.98 (2) (modified from Nesbitt, 2011: 245), ORDERED.

See comments in Nesbitt (2011: character 245).

447. Metacarpus, proximal ends: overlap (0); abut one another without overlapping (1) (Sereno & Wild, 1992; Nesbitt, 2011: 246) (Fig. 40).

See comments in Nesbitt (2011: character 246).

448. Metacarpus, width of the distal end of the metacarpal I versus its total length: 0.26–0.33 (0); 0.36–0.45 (1); 0.48–0.53 (2); 0.58–0.64 (3); 0.73–0.75 (4) (modified from Bakker & Galton, 1974; modified from Nesbitt, 2011: 251), ORDERED (Fig. 40).

See comments in Nesbitt (2011: character 251).

449. Metacarpus, extensor pits on the dorsodistal portion of the metacarpals I-III: absent or shallow and symmetrical (0); deep and asymmetrical (1) (Sereno et al., 1993; Nesbitt, 2011: 250) (Fig. 40).

See comments in Nesbitt (2011: character 250).

450. Metacarpus, metacarpal IV: longer than metacarpal III (0); equal or shorter than metacarpal III (1) (DeBraga & Rieppel, 1997; Müller, 2004: 148; Nesbitt, 2011: 260; Ezcurra, Scheyer & Butler, 2014: 204) (Fig. 40).

See comments in Nesbitt (2011: character 260).

451. Manual digits, unguals length: about the same length or shorter than the last non-ungual phalanx of the same digit (0); distinctly longer than the last non-ungual phalanx of the same digit (1) (Nesbitt et al., 2015: 222).

See comments in Nesbitt et al. (2015: character 222).

452. Manual digits, unguals of manual digits I–III: blunt on at least digits II and III (0); trenchant on digits I–III (1) (Gauthier, 1986; Nesbitt, 2011: 257) (Fig. 32 of Nesbitt, 2011).

See comments in Nesbitt (2011: character 257).

453. Manual digits, second phalanx of manual digit II: shorter than the first phalanx of manual digit II (0); longer than the first phalanx of manual digit II (1) (Gauthier, 1986; Nesbitt, 2011: 255) (Fig. 32 of Nesbitt, 2011).

See comments in Nesbitt (2011: character 255).

454. Manual digits, number of phalanges in digit IV: five (0); four (1); three or less (2) (Gauthier, 1986; Benton & Clark, 1988; Sereno, 1994; Ezcurra, Lecuona & Martinelli, 2010: 120; cf. Nesbitt, 2011: 258; Pritchard et al., 2015: 162, in part), ORDERED (Fig. 32 of Nesbitt, 2011).

See comments in Nesbitt (2011: character 258).

455. Pelvic girdle, acetabulum: completely closed (0); perforated (1) (Gauthier, 1986; Laurin, 1991: J8; Juul, 1994; Reisz, Laurin & Marjanović, 2010: 98; modified from Ezcurra, Lecuona & Martinelli, 2010: 124; Nesbitt, 2011: 273; Ezcurra, Scheyer & Butler, 2014: 96; Nesbitt et al., 2015: 223) (Figs. 9, 41, 33, 34 of Nesbitt, 2011).

See comments in Nesbitt (2011: character 273).

456. Pelvic girdle, acetabulum orientation: mainly laterally facing (0); lateroventrally or mainly ventrally facing (1) (Benton & Clark, 1988; Juul, 1994; Reisz, Laurin & Marjanović, 2010: 123; Nesbitt, 2011: 270).

See comments in Nesbitt (2011: character 270).

457. Pelvic girdle, acetabular antitrochanter: absent (0); present (1) (Sereno & Arcucci, 1994a; Ezcurra, Lecuona & Martinelli, 2010: 125; Nesbitt, 2011: 274) (Figs. 41, 33, 34 of Nesbitt, 2011).

See comments in Nesbitt (2011: character 274).

458. Ilium, maximum height of the ilium versus length of the femur: 0.12–0.17 (0); 0.21–0.47 (1); 0.54–0.57 (2) (New character), ORDERED.

This character describes the relative size of the ilium (measured as its maximum dorsoventral height) with respect to the length of the femur. In most diapsids this ratio is less than 0.5, but in the archosauromorphs Trilophosaurus buettneri (Spielmann et al., 2008) and Mesosuchus browni (SAM-PK-7416) it is greater than this value.

459. Ilium, laterally deflected iliac blade: absent (0); present (1) (Dilkes & Sues, 2009; Desojo, Ezcurra & Schultz, 2011: 96).

See comments in Desojo, Ezcurra & Schultz (2011: 860).

460. Ilium, preacetabular process: absent or incipient (0); present, being considerably anteroposteriorly shorter than its dorsoventral height (1); present, being longer than two thirds of its height and not extending beyond the level of the anterior margin of the pubic peduncle (2); present and extending beyond the level of the anterior margin of the pubic peduncle (3) (modified from Dilkes, 1998; Rieppel, Mazin & Tchernov, 1999; Müller, 2004: 67; Ezcurra, Lecuona & Martinelli, 2010: 61, in part; Nesbitt, 2011: 268 and 269; Dilkes & Arcucci, 2012: 68, in part; Ezcurra, Scheyer & Butler, 2014: 205; Pritchard et al., 2015: 169, in part), ORDERED (Figs. 9, 41, 33, 34 of Nesbitt, 2011).

See comments in Nesbitt (2011: characters 268 and 269).

461. Ilium, preacetabular process in lateral view: semicircular (0); subtriangular or finger-like (1) (reworded from Pritchard et al., 2015: 170). This character is inapplicable in taxa that lack a preacetabular process (Figs. 9 and 41).

Figure 41 Pelvic girdles of Triassic and Recent saurians in (A–E) lateral, (F) ventral, and (G) medial views.

(A) Left hemipelvis of Salvator merianae (MACN-He 47992); (B) left hemipelvis of Pamelaria dolichotrachela (ISI R316/1); (C) left hemipelvis of Prolacerta broomi (BP/1/2676); (D) right hemipelvis of Marasuchus lilloensis (PVL 3870, mirrored); (E) right ilium of Erythrosuchus africanus (NHMUK PV R3592, mirrored); (F) pelvis of Proterosuchus alexanderi (NMQR 1880); and (G) left ischium of Pamelaria dolichotrachela (ISI R316/1). Numbers indicate character-states scored in the data matrix and the arrows indicate anterior direction. Scale bars equal 5 mm in (A, C, D), 1 cm in (B, G, F), and 5 cm in (E).

The preacetabular process of the ilium has a continuously convex anterior margin and, as a result, the entire process is semi-circular in lateral view in non-archosauriform diapsids (with the exception of Macrocnemus bassanii (Rieppel, 1989a), Tanystropheus longobardicus (Nosotti, 2007) and some specimens of Jesairosaurus lehmani [ZAR 12]), proterosuchids (Proterosuchus alexanderi: NMQR 1484; “Chasmatosaurus” yuani: IVPP V4067), and species of the genus Garjainia (PIN 951/8). By contrast, the preacetabular process tapers anteriorly or is finger-like in lateral view in other archosauriforms.

462. Ilium, lateral crest dorsal to the supraacetabular crest/rim: absent (0); present and divides the preacetabular process from the postacetabular process (1); confluent with the anterior extent of the preacetabular process (2) (Nesbitt, 2011: 265) (Figs. 33 and 34 of Nesbitt, 2011).

See comments in Nesbitt (2011: character 265).

463. Ilium, length of the postacetabular process versus anteroposterior length of the acetabulum: 0.31–0.63 (0); 0.79–1.24 (1); 1.31–1.37 (2); 1.49–1.55 (3) (modified from Dilkes, 1998: 102; Müller, 2004: 113, in part; Ezcurra, Lecuona & Martinelli, 2010: 94, in part; Ezcurra, Scheyer & Butler, 2014: 206; Pritchard et al., 2015: 171), ORDERED (Fig. 41).

This character describes the relative anteroposterior elongation of the postacetabular process of the ilium with respect to the anteroposterior lenth of the iliac acetabulum. In the vast majority of species sampled here this ratio is less than 1.25, but it is higher in Tanystropheus longobardicus (Nosotti, 2007), Shansisuchus shansisuchus (Young, 1964), and Riojasuchus tenuisceps (PVL 3827, 3828).

464. Ilium, main axis of the postacetabular process in lateral or medial view: posterodorsally oriented (0); mainly posteriorly oriented (1) (Gauthier, Kluge & Rowe, 1988; Pritchard et al., 2015: 164) (Figs. 9 and 41).

The main axis of the postacetabular process of the ilium is oriented mainly posterodorsally and, as a result, the distal tip of the process is distinctly placed above the level of the preacetabular process in non-archosauromorph diapsids (with the exception of Simoedosaurus lemoinei: MNHN.F.R3313, BR12090), Protorosaurus speneri (Gottmann-Quesada & Sander, 2009), Mesosuchus browni (SAM-PK-7416), Howesia browni (NHMUK PV R5872, cast of SAM-PK-5886 that preserves portions of the ilium currently lost in the original specimen but figured by Broom (1906)), Rhynchosaurus articeps (BRLSI M20a/b), Prolacerta broomi (BP/1/2676), proterochampsids (e.g., Chanaresuchus bonapartei: MCZ 4035, PVL 6244; Pseudochampsa ischigualastensis: Trotteyn & Ezcurra, 2014), doswelliids (e.g., Archeopelta arborensis: CPEZ-239a; Doswellia kaltenbachi: USNM 244214), Turfanosuchus dabanensis (IVPP V3237), Aetosauroides scagliai (PVL 2073), and non-dinosaurian avemetatarsalians (e.g., Dimorphodon macronyx: NHMUK PV R41212-13; Lagerpeton chanarensis: PVL 4619; Marasuchus lilloensis: PVL 3870; Silesaurus opolensis: ZPAL AbIII/404/1, 907/8, 2517). The postacetabular process is mainly oriented posteriorly in other archosauromorphs scored here.

465. Ilium, caudifemoralis brevis muscle origin on the lateroventral surface of the postacetabular process: not dorsally or laterally rimed by a brevis shelf (0); dorsally rimed by a brevis shelf, but lacking a brevis fossa (1); dorsolaterally rimed by a brevis shelf and with a lateroventrally facing brevis fossa (2); laterally rimed by a brevis shelf and with a ventrally facing brevis fossa (3) (Gauthier, 1986; Juul, 1994; Ezcurra, Lecuona & Martinelli, 2010: 121, 122; Nesbitt, 2011: 271, in part) (Fig. 9).

See comments in Nesbitt (2011: character 271). Novas (1996) defined the brevis shelf as a distinct and prominent shelf on the posterolateral margin of the iliac blade, placed external to the posteroventral iliac margin, which extends from the ischiadic peduncle to the posterior end of the blade. In some archosauriforms, such as the erythrosuchid Garjainia prima (PIN 951/8), there is a distinct change in slope on the lateral surface of the postacetabular process, which is anteriorly confluent with the base of the posterior margin of the ischiadic peduncle. As a result, this change in slope is considered homologous with the posteroventral iliac margin of other diapsids rather than a brevis shelf. Although the diagonal tuberosity of tanystropheids (e.g., Tanystropheus longobardicus: Nosotti, 2007; Macrocnemus bassanii: BSPG 1973 I 86, cast of Besano II), Prolacerta broomi (BP/1/2676) and Exilisuchus tubercularis (PIN 4171/25) is not developed as a sharp shelf, this structure is topologically very similar to the brevis shelf and therefore considered as a primary homology.

466. Ilium, dorsal margin of the iliac blade: convex (0); mostly straight (1); concave (2) (modified from Dilkes, 1998; modified from Ezcurra, Lecuona & Martinelli, 2010: 60) (Fig. 41).

The dorsal margin of the iliac blade is gently convex or mostly straight in lateral view in the vast majority of diapsids. However, in Tanystropheus longobardicus (Nosotti, 2007), Parasuchus hislopi (ISI R42), Riojasuchus tenuisceps (PVL 3827, 3828), Prestosuchus chiniquensis (UFRGS-PV-0152-T), and several avemetatarsalians (e.g., Dimorphodon macronyx: NHMUK PV R41212-13; Lagerpeton chanarensis: PVL 4619; Marasuchus lilloensis: PVL 3870; Herrerasaurus ischigualastensis: PVL 2566) the dorsal margin of the iliac blade is continuously gently anteroposteriorly concave. This concavity differs from the two fairly straight margins that meet in a wide angle in some species that possess a posterodorsally oriented postacetabular process, such as Prolacerta broomi (BP/1/2676), Chanaresuchus bonapartei (MCZ 4035, PVL 6244), and Turfanosuchus dabanensis (IVPP V3237). The iliac blade of the early dinosauriform Silesaurus opolensis was originally reconstructed as having a widely concave dorsal margin, but more recently collected and more complete specimens show that this margin is mostly straight in lateral view (ZPAL AbIII/364).

467. Ilium, angle between anterior margin of the pubic peduncle and the horizontal plane of the pelvic girdle: lower than 45°(0); equal or higher than 45°(1) (Ezcurra, Lecuona & Martinelli, 2010: 95) (Fig. 9).

The pubic peduncle of most diapsids is more ventrally than anteriorly oriented in lateral view and, as a result, the angle formed by its anterior margin and the horizontal plane of the pelvic girdle is lower than 45°. The pubic peduncle is more anteriorly oriented than in most scored species in Petrolacosaurus kansensis (Reisz, 1981), Youngina capensis (BP/1/3859), Cteniogenys sp. (Evans, 1991), Macrocnemus bassanii (BSPG 1973 I 86, cast of Besano II; PIMUZ T2472, T4822), some specimens of Rhynchosaurus articeps (BRLSI M20a/b), Shansisuchus shansisuchus (Young, 1964), Dimorphodon macronyx (NHMUK PV R41212-13), and Heterodontosaurus tucki (SAM-PK-K1332). In the latter taxa, the angle formed between the anterior margin of the pubic peduncle and the horizontal plane of the pelvic girdle is equal or higher than 45°.

468. Ilium, posteriorly projected heel on the posterior margin of the ischiadic peduncle in lateral view: absent (0); present, the dorsal margin of which is set at 45°or less to the longitudinal axis of the bone (1) (Ezcurra, Lecuona & Martinelli, 2010: 96) (Fig. 41).

The presence of a posteriorly projected heel (with a dorsal margin forming an angle of 45°or less with the horizontal plane of the pelvic girdle) on the distal end of the ischiadic process of the ilium is broadly distributed among non-archosauriform diapsids and it is polymorphic for some species (e.g., Macrocnemus bassanii: BSPG 1973 I 86, cast of Besano II; PIMUZ T2472, T4822; Azendohsaurus madagaskarensis: Nesbitt et al., 2015). Within archosauriforms, this heel is present in some erythrosuchids (Garjainia prima: PIN 951/8; Garjainia madiba: BP/1/5525, 7333; Erythrosuchus africanus: NHMUK PV R3592, SAM-PK-905) and loricatans (e.g., Batrachotomus kupferzellensis: SMNS 80273; Prestosuchus chiniquensis: UFRGS-PV-0152-T). In other archosauriforms, the posterior margin of the distal end of the ischiadic peduncle is nearly vertical.

469. Ilium, acetabulum shape: irregular, marked by posterodorsal invasion by finished bone (0); roughly circular, no posterodorsal invasion by finished bone (1) (Benton, 1985; Gauthier, Kluge & Rowe, 1988; Pritchard et al., 2015: 168).

In early diapsids, such as Petrolacosaurus kansensis (Reisz, 1981), the acetabulum possesses a notched posterodorsal corner. By contrast, the posterodorsal corner of the acetabulum is continuously convex and the acetabulum as a whole acquieres a circular contour in lateral view in Acerosodontosaurus piveteaui (MNHN 1908-32-57), Youngina capensis (BP/1/3859), choristoderans (e.g., Simoedosaurus lemoinei: MNHN.F.BR12090), lepidosauromorphs (e.g., Gephyrosaurus bridensis: Evans, 1981) and archosauromorphs (e.g., Macrocnemus bassanii: BSPG 1973 I 86, cast of Besano II; PIMUZ T2472, T4822; Prolacerta broomi: BP/1/2676; Pamelaria dolichotrachela: ISI R316/1). It should be noted that the condition of Petrolacosaurus kansensis and other basal amniotes differ from the invasion of the acetabulum by an iliac antitrochanter, which possesses an articular surface and occurs in the posteroventral corner of the acetabulum of some archosauriforms (e.g., Herrerasaurus ischigualastensis: MCZ 4381).

470. Pubis-ischium, contact: present and extended ventrally (0); present and reduced to a thin proximal contact (1) (Nesbitt, 2011: 287) (Figs. 41, 33 and 36 of Nesbitt, 2011).

See comments in Nesbitt (2011: character 287).

471. Pubis-ischium, thyroid fenestra: absent (0); present (1) (Rieppel, Mazin & Tchernov, 1999; Müller, 2004: 68; Ezcurra, Scheyer & Butler, 2014: 208; Pritchard et al., 2015: 163) (Fig. 41).

The thyroid fenestra is an opening formed by the concave posterior margin of the pubis and anterior margin of the ischium and it is ventrally open because of the lack of contact between these two bones at their distal ends. This opening is present in lepidosauromorphs (e.g., Planocephalosaurus robinsonae: Fraser & Walkden, 1984; Gephyrosaurus bridensis: Evans, 1981) and tanystropheids (e.g., Amotosaurus rotfeldensis: Fraser & Rieppel, 2006; Tanystropheus longobardicus: Wild, 1973; Nosotti, 2007; Macrocnemus bassanii: Rieppel, 1989a).

472. Pubis, total length versus anteroposterior length of the acetabulum: 1.15–2.58 (0); 2.84–3.43 (1); 3.94–4.87 (2) (modified from Sereno, 1991; Juul, 1994; Weinbaum & Hungerbühler, 2007; Ezcurra, Lecuona & Martinelli, 2010: 126, 169; cf. Nesbitt, 2011: 278), ORDERED (Fig. 41).

See comments in Nesbitt (2011: character 278).

473. Pubis, anterior and posterior portions of the acetabular margin: continuous (0); recessed (1) (Sereno, 1991; Ezcurra, Lecuona & Martinelli, 2010: 127; Nesbitt, 2011: 286) (Figs. 41 and 33 of Nesbitt, 2011).

See comments in Nesbitt (2011: character 286).

474. Pubis, tuberosity for the attachment of the ambiens muscle in mature individuals: prominent (0); incipient or absent (1) (Hutchinson, 2001; Ezcurra, Lecuona & Martinelli, 2010: 62; Dilkes & Arcucci, 2012: 69) (Fig. 41).

The insertion of M. ambiens on the proximal end of the pubis occurs on a non-raised surface or a very low, rounded tuberosity in most diapsids. However, the tuberosity for the attachment of this muscle on the pubis is well laterally developed as a proximodistally elongated, mound-like structure in Petrolacosaurus kansensis (Reisz, 1981), Planocephalosaurus robinsonae (Fraser & Walkden, 1984), Pamelaria dolichotrachela (ISI R316/1), Azendohsaurus madagaskarensis (Nesbitt et al., 2015), Cuyosuchus huenei (MCNAM PV 2669), Garjainia species (BP/1/7328, PIN 951/25), and several dinosauromorphs (e.g., Lagerpeton chanarensis: PVL 4619; Marasuchus lilloensis: PVL 3870; Silesaurus opolensis: ZPAL AbIII/404/5, 1847, 3339, 3340; Herrerasaurus ischigualastensis: PVL 2566).

475. Pubis, shaft orientation: anteroventral (0); vertical or posteroventral (1) (Benton, 1985; Juul, 1994; Ezcurra, Lecuona & Martinelli, 2010: 128; Nesbitt, 2011: 279) (Fig. 33 of Nesbitt, 2011).

See comments in Nesbitt (2011: character 279).

476. Pubis, form of the shaft (=pubic tubercle, = pectineal tuberosity) in lateral view: plate-like (0); rod-like and curved posteriorly (1); rod-like and straight (2) (Ezcurra, 2006; Ezcurra, Lecuona & Martinelli, 2010: 154) (Fig. 41).

The pubic shaft is developed as a compressed, plate-like structure in the following taxa sampled here: lepidosauromorphs (Gephyrosaurus bridensis: Evans, 1981; Planocephalosaurus robinsonae: Fraser & Walkden, 1984), Simoedosaurus lemoinei (MNHN.F.BR12091), and some non-archosauriform archosauromorphs including Amotosaurus rotfeldensis (SMNS 50830), Tanystropheus longobardicus (Wild, 1973; Nosotti, 2007), Jesairosaurus lehmani (ZAR 11), and Trilophosaurus buettneri (Spielmann et al., 2008). In other diapsids, the pubic shaft is thickened with respect to the symphyseal region of the bone and acquires a rod-like morphology. The rod-like pubic shaft is posteriorly curved in several disparate archosauromorphs, such as Pamelaria dolichotrachela (ISI R316/1) and Azendohsaurus madagaskarensis (Nesbitt et al., 2015), Mesosuchus browni (SAM-PK-7416), Euparkeria capensis (SAM-PK-5867), Yarasuchus deccanensis (ISI R334), Tropidosuchus romeri (PVL 4601), and Chanaresuchus bonapartei (MCZ 4035).

477. Pubis, pubic apron: absent, symphysis extended along the ventral margin of the pelvic girdle and visible in lateral view (0); present, symphysis restricted anteriorly and obscured by the pubic shaft in lateral view (1) (Dilkes, 1998: 104; modified from Ezcurra, Scheyer & Butler, 2014: 209; Pritchard et al., 2015: 173) (Fig. 41).

The pubic shymphysis extends longitudinally along the ventral margin of the pelvic girdle in non-archosauriform diapsids (with the exception of Azendohsaurus madagaskarensis: Nesbitt et al., 2015) and proterosuchids (Proterosuchus alexanderi: NMQR 1484; “Chasmatosaurus” yuani: IVPP V4067). By contrast, the pubic shymphysis is restricted to the anterior portion of the pelvic girdle, forming a transversely broad and anteriorly facing apron in Azendohsaurus madagaskarensis (Nesbitt et al., 2015), Cuyosuchus huenei (MCNAM PV 2669), erythrosuchids (e.g., Garjainia prima: PIN 951/25; Erythrosuchus africanus: NHMUK PV R3592), Euparkeria capensis (SAM-PK-5867), Yarasuchus deccanensis (ISI R334), proterochampsids (e.g., Tropidosuchus romeri: PVL 4601; Chanaresuchus bonapartei: MCZ 4035), Doswellia kaltenbachi (USNM 244214), phytosaurs (e.g., Parasuchus hislopi: ISI R42; Smilosuchus gregorii: USNM 18313), ornithosuchids (e.g., Riojasuchus tenuisceps: PVL 3827), suchians (e.g., Turfanosuchus dabanensis: IVPP V3237; Aetosauroides scagliai: PVL 2073), and avemetatarsalians (e.g., Lagerpeton chanarensis: PVL 4619; Marasuchus lilloensis: PVL 3870; Silesaurus opolensis: ZPAL AbIII/404/5, 1847, 3339, 3340; Herrerasaurus ischigualastensis: PVL 2566).

478. Pubis, transverse width of conjoined aprons versus total length of the bone: 0.27–0.59 (0); 0.77–0.97 (1); 1.12–1.28 (2); 1.48–1.94 (3) (modified from Cooper, 1984; Ezcurra, Lecuona & Martinelli, 2010: 155), ORDERED. This character is not applicable in taxa in which the pubic apron is absent.

This character describes the relative maximum transverse width of the conjoined pubes in anterior view and the total length of the bone. The pubic apron is transversely narrow and, as a result, this ratio is less than 0.60 in ornithosuchids (e.g., Ornithosuchus longidens: Walker, 1964; Riojasuchus tenuisceps: PVL 3827), gracilisuchids (e.g., Turfanosuchus dabanensis: IVPP V3237; Gracilisuchus stipanicicorum: PVL 4597), loricatans (e.g., Batrachotomus kupferzellensis: SMNS 80269; Prestosuchus chiniquensis: UFRGS-PV-0152-T), and dinosauriforms (e.g., Marasuchus lilloensis: PVL 3870; Silesaurus opolensis: ZPAL AbIII/404/5; Herrerasaurus ischigualastensis: PVL 2566).

479. Pubis, pectineal process: absent (0); present (1) (Ezcurra, Scheyer & Butler, 2014: 207; Pritchard et al., 2015: 175) (Fig. 41).

The pectineal process is an anterolaterally or anteriorly directed, short tuberosity on the pubic shaft of Petrolacosaurus kansensis (Reisz, 1981), Gephyrosaurus bridensis (Evans, 1981), and Trilophosaurus buettneri (Spielmann et al., 2008).

480. Pubis, distal end in lateral or medial view: unexpanded or gently expanded anteroposteriorly (0); sharply expanded anteroposteriorly, forming a distinct pubic boot (1) (Gauthier, 1986; Nesbitt, 2011: 283). This character is inapplicable in taxa that lack a rod-like pubic shaft (Fig. 33 of Nesbitt, 2011).

See comments in Nesbitt (2011: character 283).

481. Pubis, transverse width of the distal portion: nearly as broad as the proximal width (0); significantly narrower than the proximal width (1) (Galton, 1976; Nesbitt, 2011: 289). This character is inapplicable in taxa that lack a rod-like pubic shaft.

See comments in Nesbitt (2011: character 289).

482. Ischium, total length versus anteroposterior length of the acetabulum: 1.04–1.24 (0); 1.55–2.50 (1); 2.72–3.53 (2); 4.31–4.48 (3) (Benton, 2004; modified from Ezcurra, Lecuona & Martinelli, 2010: 63; cf. Nesbitt, 2011: 282, 298; Dilkes & Arcucci, 2012: cf. 70; Nesbitt et al., 2015: 225), ORDERED.

See comments in Nesbitt (2011: characters 282 and 298).

483. Ischium, proximal articular surface: articular surface with the ilium and pubis continuous (0); articular surfaces with the ilium and pubis separated by a fossa (1); articular surfaces with the ilium and pubis separated by a non-articulating notched surface (2) (Irmis et al., 2007a; Nesbitt, 2011: 297), ORDERED (Figs. 41, 33, 36 of Nesbitt, 2011).

See comments in Nesbitt (2011: character 297).

484. Ischium, longitudinal groove on the dorsal surface of shaft: absent (0); present (1) (Yates, 2003).

The dorsal surface of the ischium possesses a longitudinal groove in Youngina capensis (BP/1/3859), Silesaurus opolensis (ZPAL AbIII/404/1, 1228), and Herrerasaurus ischigualastensis (MCZ 4381) among the diapsids sampled here. In other taxa, the dorsal surface of the ischium is continuously convex transversely.

485. Ischium, medial contact with antimere: restricted to the medial edge (0); extensive contact but the dorsal margins are separated (1) (Nesbitt, 2011: 291) (Fig. 36 of Nesbitt, 2011).

See comments in Nesbitt (2011: character 291).

486. Ischium, symphysis raised on a distinct low peduncle: absent (0); present (1) (New character) (Fig. 41).

The symphyseal facet of the ischium is placed at the same level as the rest of the medial surface of the bone in the vast majority of diapsids. However, this facet is raised on a low peduncle in Pamelaria dolichotrachela (ISI R316/1) and Azendohsaurus madagaskarensis (Nesbitt et al., 2015).

487. Ischium, cross-section of the distal portion: plate-like (0); semicircular or subtriangular (1) (Sereno, 1999; modified from Nesbitt, 2011: 293) (Fig. 36 of Nesbitt, 2011).

See comments in Nesbitt (2011: character 293).

488. Ischium, shape of posterior margin: linear posterior margin (0); posterior process extends from posterodorsal ischiadic margin (1) (Merck, 1997; Pritchard et al., 2015: 176) (Fig. 41).

The posterior margin of the ischium is rounded or tapers in the vast majority of diapsids, but it possesses a posteriorly projecting posterodorsal process in the early rhynchocephalian Planocephalosaurus robinsonae (Fraser & Walkden, 1984) and the non-archosauriform archosauromorphs Protorosaurus speneri (SMNS 55387, cast of Simon/Bartholomäus specimen), Amotosaurus rotfeldensis (SMNS 50830), Macrocnemus bassanii (BSPG 1973 I 86, cast of Besano II; PIMUZ T2472), Tanystropheus longobardicus (Wild, 1973; Nosotti, 2007), and Prolacerta broomi (BP/1/2676).

489. Femur, total length versus total length of the humerus: 0.92–0.97 (0); 1.09–1.56 (1); 1.62–1.74 (2); 1.86–1.96 (3) (Laurin, 1991: C3; Reisz, Laurin & Marjanović, 2010: 104; cf. Ezcurra, Lecuona & Martinelli, 2010: 59; cf. Nesbitt, 2011: 231; modified from Ezcurra, Scheyer & Butler, 2014: 103), ORDERED (Fig. 15).

See comments in Nesbitt (2011: character 231).

490. Femur, minimum transverse width versus minimum transverse width of the humerus: 0.95–1.01 (0); 1.08–1.32 (1); 1.46–1.80 (2); 1.93–2.00 (3) (DeBraga & Reisz, 1995: 38; Reisz, Laurin & Marjanović, 2010: 105; modified from Ezcurra, Scheyer & Butler, 2014: 104), ORDERED (Fig. 15).

The minimum transverse width of the femur is similar to that of the humerus in most diapsids (e.g., ratio 0.95–1.35). By contrast, the femoral shaft is considerably broader than that of the humerus (i.e., >1.40) in some specimens of Macrocnemus bassanii (PIMUZ T2472), Rhynchosaurus articeps (NHMUK PV R1239), Boreopricea funerea (Benton & Allen, 1997: Fig. 2), some specimens of Euparkeria capensis (SAM-PK-5867), rhadinosuchine proterochampsids (Gualosuchus reigi: PVL 4576; Chanaresuchus bonapartei: MCZ 4035, PVL 4575), Jaxtasuchus salomoni (SMNS 91002), Riojasuchus tenuisceps (PVL 3828), and several suchians (e.g., Turfanosuchus dabanensis: IVPP V3237; Aetosauroides scagliai: PVL 2052, 2073) and Marasuchus lilloensis (PVL 3871).

491. Femur, proximal articular surface: well ossified, being flat or convex (0); partially ossified, being concave and sometimes with a circular pit (1) (Pritchard et al., 2015: 178) (Fig. 42).

Figure 42 Femora of Triassic archosauromorphs in (A–H) proximal and (I–L) distal views.

(A) Left element of Pamelaria dolichotrachela (ISI R316/1, mirrored); (B, J) left element of Erythrosuchus africanus (NHMUK R3592, mirrored); (C) right element of Dorosuchus neoetus (PIN 1579/61-2); (D) right element of Yarasuchus deccanensis (ISI Runnumbered); (E) right element of Chanaresuchus bonapartei (MCZ 4035); (F) left element of Parasuchus hislopi (ISI R42, mirrored); (G, L) left element of Silesaurus opolensis (ZPAL AbIII/361-25, mirrored); (H) right element of Herrerasaurus ischigualastensis (PVL 2566); (I) left element of Proterosuchus fergusi (SAM-PK-K140, mirrored); and (K) right element of Lagerpeton chanarensis (MCZ 4121). Numbers indicate character-states scored in the data matrix and the arrows indicate anterior/dorsal direction. Not to scale.

See comments in Pritchard et al. (2015: character 178).

492. Femur, femoral head: not distinctly offset from the shaft (0); distinctly offset from the shaft (1) (Gauthier, 1986; Juul, 1994; Ezcurra, Lecuona & Martinelli, 2010: 129).

The femoral head is not distinctly offset from the rest of the bone, being confluent with the shaft or poorly differentiated in all non-archosaurian diapsids scored here and in most pseudosuchians. In Riojasuchus tenuisceps (PVL 3827, 3828), Dimorphodon macronyx (NHMUK PV R41212-13), Lagerpeton chanarensis (PULR 06, PVL 4619), Heterodontosaurus tucki (SAM-PK-K1332) and Herrerasaurus ischigualastensis (MACN-Pv 18060, PVL 2566, PVSJ 373) the femoral head is distinctly offset from the shaft, forming a distinct neck between them and lending a hook-shaped outline to the proximal half of the bone in anterior/dorsal or posterior/ventral view.

493. Femur, femoral head orientation (long axis of the femoral head angle with respect to the transverse axis through the femoral condyles Parrish, 1986): anterior (60–90°) (0); anteromedial (20–60°) (1) (Benton & Clark, 1988; Nesbitt, 2011: 305; Dilkes & Arcucci, 2012: 75) (Fig. 43).

Figure 43 Femora of Triassic archosauromorphs in (A, C, E, G, I–K) medial and (B, D, F, H, L) posterior/ventral views.

(A, B) Left element of Tanystropheus longobardicus (SMNS unnumbered); (C; D) left element of Prolacerta broomi (BP/1/2676); (E, F) left element of Proterosuchus fergusi (SAM-PK-K140); (G, H) left element of Erythrosuchus africanus (NHMUK PV R3592); (I) right element of Chanaresuchus bonapartei (MCZ 4035); (J) right element of Jaxtasuchus salomoni (SMNS 91002); and (K, L) left element of Aetosauroides scagliai (PVL 2073, [K] mirrored). Numbers indicate character-states scored in the data matrix and the arrows indicate anterior/dorsal direction. Scale bars equal 2 cm in (A, B, E, F, K, L), 5 mm in (C, D), 5 cm in (G, H), and 1 cm in (I, J).

See comments in Nesbitt (2011: character 305).

494. Femur, proximal articular surface (=posterolateral portion of the head sensu Nesbitt, 2011): limited to the proximal surface of the bone (0); extends under the proximal surface of the bone (1) (reworded from Sereno & Arcucci, 1994a; Ezcurra, Lecuona & Martinelli, 2010: 130; Nesbitt, 2011: 313) (Figs. 38 and 39 of Nesbitt, 2011).

See comments in Nesbitt (2011: character 313).

495. Femur, proximal surface: rounded and smooth (0); transverse groove present (1) (modified from Ezcurra, 2006; modified from Nesbitt, 2011: 314) (Figs. 42, 38 and 39 of Nesbitt, 2011).

See comments in Nesbitt (2011: character 314). This character is independent from character 491 because there are taxa with a well-ossified femoral head that possess a transverse groove on its proximal surface (e.g., Ornithosuchus longidens: Walker, 1964; Silesaurus opolensis: Dzik, 2003).

496. Femur, posteromedial tuber (=anteromedial tuber of Nesbitt, 2011) on the femoral head: absent (0); present (1) (Gauthier, 1986; reworded from Nesbitt, 2011: 300, in part; Dilkes & Arcucci, 2012: 73) (Fig. 42).

See comments in Nesbitt (2011: character 300). The “anteromedial tuber” of Nesbitt (2011) is here renamed posteromedial tuber to follow conventional archosaur anatomical nomenclature.

497. Femur, posterior tuber (=posteromedial tuber of Nesbitt, 2011) on the femoral head: present (0); absent (1) (Novas, 1996; Ezcurra, Lecuona & Martinelli, 2010: 156; reworded from Nesbitt, 2011: 301, in part; Dilkes & Arcucci, 2012: 74). This character is inapplicable in taxa in which the internal trochanter reaches the proximal margin of the bone (Fig. 42).

See comments in Nesbitt (2011: character 301). The “posteromedial tuber” of Nesbitt (2011) is here renamed posterior tuber following the original terminology used by Novas (1996) and to be consistent with conventional archosaur anatomical nomenclature.

498. Femur, anterior tuber (=anterolateral tuber of Nesbitt, 2011) on the femoral head: present as an expansion (0); absent (1) (Sereno & Arcucci, 1994a; reworded from Nesbitt, 2011: 302) (Fig. 42).

See comments in Nesbitt (2011: character 302). The “anterolateral tuber” of Nesbitt (2011) is here renamed anterior tuber to follow conventional archosaur anatomical nomenclature.

499. Femur, fossa trochanterica (sensu Novas, 1996) on the ventral/posterior surface of the proximal end: present (0); absent (1) (modified from Novas, 1996; Benton, 2004; Ezcurra, Lecuona & Martinelli, 2010: 65,131; Dilkes & Arcucci, 2012: 72) (Fig. 42).

The proximal articular surface of the femoral head is slightly depressed and concave on its posterolateral corner, immediately lateral of the posterior tuber (if present), in dinosauriform archosaurs such as Marasuchus lilloensis (PVL 3870, 3871), Silesaurus opolensis (ZPAL AbIII/361/25, 2063), and Herrerasaurus ischigualastensis (MACN-Pv 18060, PVL 2566, PVSJ 373). This depression has been called fossa trochanterica or facies articularis antitrochanterica in recent contributions (Novas, 1996; Langer, 2003; Langer & Benton, 2006). In other diapsids, the posterolateral corner of the femoral head is convex.

500. Femur, dorsolateral trochanter on the anterolateral surface of the proximal end: absent (0); present (1) (Chatterjee, 1987; Nesbitt, 2011: 307). This character is inapplicable in taxa with a wing-like anterior trochanter that extends proximally close to the greater trochanter (Fig. 38 of Nesbitt, 2011).

See comments in Nesbitt (2011: character 307).

501. Femur, transition between femoral head and shaft: smooth (0); notch (1); concave emargination (2) (Sereno & Arcucci, 1994a; Novas, 1996; Nesbitt, 2011: 304) (Fig. 39 of Nesbitt, 2011).

See comments in Nesbitt (2011: character 304).

502. Femur, anterior trochanter (=lesser or minor trochanter, = iliofemoralis cranialis muscle insertion): absent (0); present (1) (Gauthier, 1986; Novas, 1992b; Juul, 1994; Ezcurra, Lecuona & Martinelli, 2010: 132; Nesbitt, 2011: 308, in part; Dilkes & Arcucci, 2012: 76) (Figs. 37 and 39 of Nesbitt, 2011).

See comments in Nesbitt (2011: character 308).

503. Femur, trochanteric shelf (=iliofemoralis externus muscle insertion): absent (0); present in mature individuals (1) (Gauthier, 1986; Novas, 1996; Ezcurra, Lecuona & Martinelli, 2010: 133; Nesbitt, 2011: 311; Nesbitt et al., 2015: 227).

See comments in Nesbitt (2011: character 311).

504. Femur, attachment of the caudifemoralis musculature on the posterior surface of the bone: crest-like and with intertrochanteric fossa (=internal trochanter), and convergent with proximal end (0); crest-like and with intertrochanteric fossa (=internal trochanter), and not convergent with proximal end (1); crest-like and without intertrochanteric fossa (=fourth trochanter), and not convergent with proximal end (2) (Gauthier, Kluge & Rowe, 1988; modified from Juul, 1994; Ezcurra, Lecuona & Martinelli, 2010: 64; Nesbitt, 2011: 315 and 316, in part; Dilkes & Arcucci, 2012: 71, in part; Pritchard et al., 2015: 179, in part; Nesbitt et al., 2015: 226, in part), ORDERED. Scored as inapplicable in taxa without a distinct process for the attachment of the caudifemoralis musculature (Figs. 42 and 43).

It is followed here Nesbitt et al. (2009) regarding the hypothesis of homology between the internal and fourth trochanter, which relies on the interpretation that both structures represent the attachment area of M. caudifemoralis, are placed in the same area of the femoral shaft, and they are not present simultaneously in any saurian species. The internal trochanter is a crest-like process that defines a broad intertrochanteric fossa, whereas the fourth trochanter lacks such a fossa and never reaches the proximal end of the femur. The internal trochanter reaches the proximal end of the femur in most non-archosauriform diapsids (e.g., Youngina capensis: BP/1/3859; Planocephalosaurus robinsonae: (Fraser & Walkden, 1984); Tanystropheus longobardicus: SMNS unnumbered; Pamelaria dolichotrachela: ISI R316/1; Noteosuchus colletti: Carroll, 1976; Prolacerta broomi: BP/1/2676), proterosuchids (e.g., Proterosuchus fergusi: SAM-PK-K140; Proterosuchus alexanderi: NMQR 1484; “Chasmatosaurus” yuani: IVPP V2719), Garjainia prima (PIN 951/27) and Garjainia madiba (BP/1/5767). The condition in which the internal trochanter does not reach the proximal end of the femur is therefore interpreted as intermediate to the presence of a fourth trochanter and, as a result, the character is considered additive. A M. caudifemoralis attachment process that is not convergent with the proximal end of the femur and with an intertrochanteric fossa (i.e., internal trochanter) is present in Petrolacosaurus kansensis (Reisz, 1981), Gephyrosaurus bridensis (Evans, 1981), Simoedosaurus lemoinei (MNHN.F.BR1348, R3404), Azendohsaurus madagaskarensis (Nesbitt et al., 2015), Trilophosaurus buettneri (Spielmann et al., 2008) and Erythrosuchus africanus (NHMUK PV R3592). A fourth trochanter is present in Shansisuchus shansisuchus (Young, 1964; D Gower, pers. comm., 2015), Euparkeria capensis (SAM-PK-5883), Dorosuchus neoetus (PIN 1579/61), Yarasuchus deccanensis (ISI R334), Dongusuchus efremovi (PIN 952/15-1–5), proterochampsids (e.g., Tropidosuchus romeri: PVL 4601, 4604; Gualosuchus reigi: PULR 05, PVL 4576; Chanaresuchus bonapartei: MCZ 4035, PVL 4575, 6244), doswelliids (e.g., Archeopelta arborensis: CPEZ-239a; Jaxtasuchus salomoni: SMNS 91002), and more crownward archosauriforms (e.g., Parasuchus hislopi: ISI R42; Smilosuchus gregorii: USNM 18313; Riojasuchus tenuisceps: PVL 3827, 3828; Aetosauroides scagliai: PVL 2052, 2073; Lagerpeton chanarensis: PULR 06, PVL 4619; Heterodontosaurus tucki: SAM-PK-K1332; Herrerasaurus ischigualastensis: MACN-Pv 18060, PVL 2566, PVSJ 373).

505. Femur, shape of the process for the attachment of the caudifemoralis musculature: mound-like and rounded (0); sharp flange (1) (Gauthier, 1986; modified from Nesbitt, 2011: 316). Scored as inapplicable in taxa without a distinct process for the attachment of the caudifemoralis musculature (Figs. 37–39 of Nesbitt, 2011).

See comments in Nesbitt (2011: character 316).

506. Femur, process for the attachment of the caudifemoralis musculature in medial or lateral view: symmetrical, with the proximal and distal margins forming similar low-angle slopes to the shaft (0); asymmetrical, with the distal margin forming a steeper angle to the shaft (1) (Langer & Benton, 2006; Nesbitt, 2011: 317). Scored as inapplicable in taxa without a distinct process for the attachment of the caudifemoralis musculature (Figs. 38 and 39 of Nesbitt, 2011).

See comments in Nesbitt (2011: character 317).

507. Femur, proximodistal extension of the process for the attachment of the caudifemoralis musculature: restricted to the proximal half of the shaft and low (0); distally extended beyond mid-shaft and well posteriorly developed (1) (New character). Scored as inapplicable in taxa without a distinct process for the attachment of the caudifemoralis musculature (Fig. 43).

The attachment of the caudifemoralis musculature is restricted to the proximal half of the femur in most diapsids, but in the rhadinosuchine proterochampsids Gualosuchus reigi (PULR 05, PVL 4576) and Chanaresuchus bonapartei (MCZ 4035, PVL 4575, 6244) this process is distally extended beyond the mid-shaft of the bone and is strongly posteriorly developed.

508. Femur, bone wall thickness at or near midshaft: thickness/diameter >0.3 (0); thin, thickness/diameter <0.3 (1) (Nesbitt, 2011: 323).

See comments in Nesbitt (2011: character 323).

509. Femur, shaft: diameter constant or widening distally (0); diameter distally narrowed (1) (Senter, 2004: 61; Ezcurra, Scheyer & Butler, 2014: 210) (Fig. 43).

Senter (2004) recognized the presence of a femoral shaft that becomes anteroposteriorly/dorsoventrally narrower towards the distal end of the bone as an apomorphic feature of prolacertiform archosauromorphs. This condition is here recognized only in a few early archosauromorphs, namely Protorosaurus speneri (SMNS 55387, cast of Simon/Bartholomäus specimen) and the tanystropheids Amotosaurus rotfeldensis (SMNS 54810), Macrocnemus bassanii (BSPG 1973 I 86, cast of Besano II; PIMUZ T2472, T4355, T4822), and Tanystropheus longobardicus (SMNS 54626, 59380, unnumbered).

510. Femur, distal transverse width versus total length: 0.08–0.11 (0); 0.13–0.24 (1); 0.26–0.36 (2); 0.39–0.41 (3) (Reisz & Dilkes, 2003: 21; Reisz, Laurin & Marjanović, 2010: 103; modified from Ezcurra, Scheyer & Butler, 2014: 102), ORDERED (Fig. 43).

This character describes the relative slenderness of the femur using the ratio between the maximum transverse width of the distal end and the total length of the bone.

511. Femur, distal condyles: prominent, strong dorsoventral expansion (in sprawling orientation) restricted to the distal end (0); not projecting markedly beyond shaft and expand gradually if there is any expansion (1) (Gauthier, Kluge & Rowe, 1988; Nesbitt, 2011: 318; Dilkes & Arcucci, 2012: 77; Pritchard et al., 2015: 181) (Fig. 43).

See comments in Nesbitt (2011: character 318).

512. Femur, distal articular surface: uneven, lateral (=fibular) condyle projecting distally distinctly beyond medial (=tibial) condyle (0); both condyles prominent distally and approximately at same level (1); both condyles do not project distally (distal articular surface concave or almost flat) (2) (Gauthier, Kluge & Rowe, 1988; Laurin, 1991: B8; modified from Rieppel, Mazin & Tchernov, 1999; Müller, 2004: 72; Reisz, Laurin & Marjanović, 2010: 102; Ezcurra, Lecuona & Martinelli, 2010: 160, in part; Ezcurra, Scheyer & Butler, 2014: 101) (Figs. 42 and 43).

The distal articular surface of the femur possesses a more distally projecting fibular condyle than the tibial one in Petrolacosaurus kansensis (Reisz, 1981), allokotosaurians (Pamelaria dolichotrachela: ISI R316/1; Azendohsaurus madagaskarensis: Nesbitt et al., 2015; Trilophosaurus buettneri: Spielmann et al., 2008), Prolacerta broomi (BP/1/2676), Jaxtasuchus salomoni (SMNS 91002), phytosaurs (e.g., Parasuchus hislopi: ISI R42; Smilosuchus gregorii: USNM 18313), ornithosuchids (e.g., Riojasuchus tenuisceps: PVL 3827, 3828), Nundasuchus songeaensis (Nesbitt et al., 2014), and most suchians (e.g., Turfanosuchus dabanensis: IVPP V3237; Aetosauroides scagliai: PVL 2073; Prestosuchus chiniquensis: UFRGS-PV-0152-T). In other diapsids scored here the distal condyles of the femur are similarly developed distally, with the exception of Garjainia prima (PIN 951/27), Erythrosuchus africanus (NHMUK PV R3592), Yarasuchus deccanensis (ISI R334), Dongusuchus efremovi (PIN 952/15-1–5), and Proterochampsa barrionuevoi (PVSJ 606), in which the distal articular surface of the bone is unossified and is concave or almost flat.

513. Femur, anterior extensor groove: absent, anterior margin of the bone straight or convex in distal view (0); present, anterior margin of the bone concave in distal view (1) (Rauhut, 2003) (Fig. 42).

The anterior surface of the distal end of the femur of the vast majority of diapsids sampled here possesses a longitudinal, transversely broad fossa that opens distally and partially delimits the tibial and fibular condyles by a transversely concave margin in distal view. By contrast, this extensor fossa is absent and the anterior margin of the femur is continuously transversely convex or straight in distal view in Simoedosaurus lemoinei (MNHN.F.R3404, BR1348), Trilophosaurus buettneri (Spielmann et al., 2008), Euparkeria capensis (SAM-PK-5883), Riojasuchus tenuisceps (PVL 3827, 3828), and the dinosauriforms Marasuchus lilloensis (PVL 3870, 3871), Silesaurus opolensis (ZPAL AbIII/361/25, 2063), Herrerasaurus ischigualastensis (MACN-Pv 18060, PVL 2566, PVSJ 373), and Heterodontosaurus tucki (SAM-PK-K1332).

514. Femur, surface between the lateral (=fibular) condyle and crista tibiofibularis on the distal surface: smooth (0); deep groove (1) (Rowe, 1989; Nesbitt, 2011: 322) (Fig. 42, Figs. 37–39 of Nesbitt, 2011).

See comments in Nesbitt (2011: character 322).

515. Femur, shape of lateral (=fibular) condyle in distal view: lateral surface is rounded and mound-like (0); lateral surface is triangular and sharply pointed (1) (reworded from Pritchard et al., 2015: 182) (Fig. 42).

The lateral margin of the lateral condyle of the distal end of the femur is continuously anteroposteriorly convex in distal view in Youngina capensis (BP/1/3859), erythrosuchids (Garjainia prima: PIN 951/27; Garjainia madiba: BP/1/5767; Erythrosuchus africanus: NHMUK PV R3592; Shansisuchus shansisuchus: Young, 1964), Cuyosuchus huenei (MCNAM PV 2669), Vancleavea campi (AMNH 30884 cast), Euparkeria capensis (SAM-PK-5883), Yarasuchus deccanensis (ISI R334), Dongusuchus efremovi (PIN 952/15-1, 2), most proterochampsids (e.g., Proterochampsa barrionuevoi: PVSJ 606; Tropidosuchus romeri: PVL 4601; Chanaresuchus bonapartei: PVL 4575), doswelliids (e.g., Archeopelta arborensis: CPEZ-239a; Jaxtasuchus salomoni: SMNS 91002), phytosaurs (e.g., Parasuchus hislopi: ISI R42; Smilosuchus gregorii: USNM 18313), ornithosuchids (e.g., Riojasuchus tenuisceps: PVL 3827, 3828), Nundasuchus songeaensis (Nesbitt et al., 2014), suchians (e.g., Turfanosuchus dabanensis: IVPP V3237; Aetosauroides scagliai: PVL 2073; Prestosuchus chiniquensis: UFRGS-PV-0152-T), and avemetatarsalians (e.g., Lagerpeton chanarensis: PULR 06, PVL 4619; Silesaurus opolensis: ZPAL AbIII/361/25; Heterodontosaurus tucki: SAM-PK-K1332; Herrerasaurus ischigualastensis: MACN-Pv 18060, PVL 2566, PVSJ 373). By contrast, this margin possesses a sharply laterally pointed apex in Simoedosaurus lemoinei (MNHN.F.BR1348), non-archosauriform archosauromorphs (e.g., Tanystropheus longobardicus: SMNS 54626; Pamelaria dolichotrachela: ISI R316/1; Mesosuchus browni: SAM-PK-7416; Prolacerta broomi: BP/1/2676) and proterosuchids (e.g., Proterosuchus fergusi: SAM-PK-K140; “Chasmatosaurus” yuani: IVPP V2719, V4067).

516. Tibia, total length versus total length of the femur: 0.46–0.51 (0); 0.60–0.65 (1); 0.70–1.27 (2); 1.41–1.46 (3) (Benton, 2004; modified from Ezcurra, Lecuona & Martinelli, 2010: 66; modified from Nesbitt, 2011: 299), ORDERED (Fig. 15).

See comments in Nesbitt (2011: character 299).

517. Tibia, distinctly anteriorly projected process beyond the articular portion for the femur on the proximal end (=cnemial crest): absent (0); present and anteriorly straight (1); present and curved anterolaterally (2) (Gauthier, 1986; Benton & Clark, 1988; Juul, 1994; Ezcurra, Lecuona & Martinelli, 2010: 134, in part; Nesbitt, 2011: 328) (Fig. 44).

Figure 44 Posterior zeugopodia of Triassic archosauriforms in (A–C) proximal, (D) medial, (E–G) lateral, and (H–J) distal views.

(A, D, H) Right tibia of “Chasmatosaurus” yuani (IVPP V2719, mirrored); (B, E, I) left tibia of Chanaresuchus bonapartei (PVL 4575); (C, F, J) right tibia of Silesaurus opolensis (ZPAL AbIII/403-2, 4, mirrored); and (G) left fibula of Aetosauroides scagliai (PVL 2073). Numbers indicate character-states scored in the data matrix and the arrows indicate anterior direction. Scale bars equal 1 cm.

See comments in Nesbitt (2011: character 328).

518. Tibia, proximal surface of the lateral condyle: convex or flat (0); depressed (1) (Nesbitt, 2011: 330; Dilkes & Arcucci, 2012: 80) (Fig. 40 of Nesbitt, 2011).

See comments in Nesbitt (2011: character 330).

519. Tibia, lateral posterior condyle of the proximal end: offset anteriorly from the medial posterior condyle (0); level with the medial posterior condyle at its posterior border (1) (Langer & Benton, 2006; Nesbitt, 2011: 331) (Fig. 44).

See comments in Nesbitt (2011: character 331).

520. Tibia, lateral surface of the proximal half: smooth (0); with a longitudinal crest (=fibular crest) (1) (Gauthier, 1986; Nesbitt, 2011: 333) (Fig. 44).

See comments in Nesbitt (2011: character 333).

521. Tibia, posterolateral process (=lateral malleolus) in the distal end: absent (0); present (1) (Novas, 1992a; Juul, 1994; reworded from Ezcurra, Lecuona & Martinelli, 2010: 135; Nesbitt, 2011: 334). This character is inapplicable if the distal ends of tibia and fibula are fused to each other and the profile of the posterolateral process of the tibia cannot be determined (Fig. 44).

See comments in Nesbitt (2011: character 334).

522. Tibia, posterior surface of the distal end: rounded (0); distinct proximodistally oriented ridge present (1) (Nesbitt, 2011: 336) (Fig. 40 of Nesbitt, 2011).

See comments in Nesbitt (2011: character 336).

523. Tibia, posterior side of the distal portion: smooth and featureless (0); dorsoventrally oriented groove or gap (1) (Nesbitt, 2011: 337; Dilkes & Arcucci, 2012: 78) (Fig. 40 of Nesbitt, 2011).

See comments in Nesbitt (2011: character 337).

524. Tibia, lateral side of the distal portion: smooth/rounded (0); proximodistally oriented groove (1) (Novas, 1996; Nesbitt, 2011: 338; Dilkes & Arcucci, 2012: 79). This character is inapplicable if the distal ends of tibia and fibula are fused to each other (Fig. 44).

See comments in Nesbitt (2011: character 338).

525. Fibula, proximal end in proximal view: round or slightly elliptical (0); transversely compressed (1) (Nesbitt, 2011: 341) (Fig. 41 of Nesbitt, 2011).

See comments in Nesbitt (2011: character 341).

526. Fibula, anterior edge of the proximal portion: rounded (0); tapers to a point and arched anteromedially (1) (Nesbitt, 2011: 342) (Fig. 41 of Nesbitt, 2011).

See comments in Nesbitt (2011: character 342).

527. Fibula, proximal portion in lateral view: symmetrical or nearly symmetrical (0); posterior part expanded posteriorly (1) (Nesbitt, 2011: 343) (Fig. 44, Fig. 41 of Nesbitt, 2011).

See comments in Nesbitt (2011: character 343).

528. Fibula, transverse width at mid-length: subequal to transverse width of the tibia (0); distinctly narrower than transverse width of the tibia (1) (Gauthier, 1986; Juul, 1994; reworded from Ezcurra, Lecuona & Martinelli, 2010: 136) (Fig. 15).

In non-archosauromorph diapsids, the transverse width of the tibial and fibular shafts at their mid-lengths is subequal (e.g., Petrolacosaurus kansensis: Reisz, 1981; Youngina capensis: BP/1/3859; Gephyrosaurus bridensis: Evans, 1981). In the vast majority of archosauromorphs sampled here, the fibular shaft is distinctly narrower than that of the tibia at about mid-length, but an exception to this condition occurs in the tanystropheid Tanystropheus longobardicus (PIMUZ T2817).

529. Fibula, area of attachment of the iliofibularis muscle: not on a prominent process (0); on a low, distinct tubercle (1); on a hypertrophied tubercle (2) (modified from Benton, 2004; modified from Ezcurra, Lecuona & Martinelli, 2010: 67; Nesbitt, 2011: 339, in part), ORDERED (Fig. 44, Fig. 41 of Nesbitt, 2011).

See comments in Nesbitt (2011: character 339).

530. Fibula, location of the attachment site of the iliofibularis muscle: near the proximal portion (0); near the midpoint between the proximal and distal ends (1) (Sereno, 1991; Nesbitt, 2011: 340) (Fig. 44, Fig. 41 of Nesbitt, 2011).

See comments in Nesbitt (2011: character 340).

531. Fibula, distal end in lateral view: angled anterodorsally (asymmetrical) (0); rounded or flat (symmetrical) (1) (Nesbitt, 2011: 345) (Fig. 44, Fig. 41 of Nesbitt, 2011).

See comments in Nesbitt (2011: character 345).

532. Proximal tarsals, articulation between astragalus and calcaneum: roughly flat (0); concavoconvex with concavity on the calcaneum (1); concavoconvex with concavity on the astragalus (2); fused (3) (Laurin, 1991: F8; Sereno, 1991; Müller, 2004: 171; Reisz, Laurin & Marjanović, 2010: 107; Nesbitt, 2011: 368 and 370; Ezcurra, Scheyer & Butler, 2014: 106, in part; Pritchard et al., 2015: 185, in part) (Fig. 45).

Figure 45 Proximal tarsals of Triassic archosauromorphs in (A, B, E, F) anterior/dorsal and (C, D, G, H) proximal views.

(A) Macrocnemus bassanii (PIMUZ T4822); (B, D) Proterosuchus alexanderi (NMQR 1484); (C, E) Euparkeria capensis (UMZC T692); (F, H) Smilosuchus gregorii (USNM 18313); and (G) Aetosauroides scagliai (PVL 2052). Numbers indicate character-states scored in the data matrix and the arrows indicate anterior direction. Scale bars equal 1mm (A, C, E), 5 mm (B, D), and 1 cm (F–H).

See comments in Nesbitt (2011: characters 368 and 370).

533. Proximal tarsals, foramen for the passage of the perforating artery between the astragalus and calcaneum (=perforating foramen): present (0); absent (1) (Rieppel, Mazin & Tchernov, 1999; Müller, 2004: 74; Ezcurra, Lecuona & Martinelli, 2010: 68; Nesbitt, 2011: 369; Dilkes & Arcucci, 2012: 81; Ezcurra, Scheyer & Butler, 2014: 212; Pritchard et al., 2015: 186) (Fig. 45).

See comments in Nesbitt (2011: character 369).

534. Astragalus, crural facets: separated by a non-articular surface (0); continuous (1) (Sereno & Arcucci, 1990; Sereno, 1991; Ezcurra, Lecuona & Martinelli, 2010: 69; Nesbitt, 2011: 365; Dilkes & Arcucci, 2012: 82; Nesbitt et al., 2015: 228) (Fig. 45).

See comments in Nesbitt (2011: character 365).

535. Astragalus, margin between tibial and fibular facets: grades smoothly into anterior hollow (0); separated by a prominent ridge from anterior hollow (1) (reworded from Nesbitt et al., 2015: 239). This character is inapplicable in taxa with confluent tibial and fibular facets.

See comments in Nesbitt et al. (2015: character 239).

536. Astragalus, tibial facet: concave, flat or flexed (0); divided into distinct posteromedial and anterolateral basins (1) (Sereno, 1991; Ezcurra, Lecuona & Martinelli, 2010: 73; Nesbitt, 2011: 366; Dilkes & Arcucci, 2012: 86) (Fig. 45).

See comments in Nesbitt (2011: character 366).

537. Astragalus, ascending process (=anterior ascending process): absent (0); present, occupying most of the anteroposterior depth of the astragalus (1); present, restricted to the anterior half of the astragalar depth (2) (Gauthier, 1986; Novas, 1989; Ezcurra, Lecuona & Martinelli, 2010: 137; Nesbitt, 2011: 356) ORDERED (Fig. 46 of Nesbitt, 2011).

See comments in Nesbitt (2011: character 356).

538. Astragalus, anterior hollow: shallow depression (0); reduced to a foramen (=extensor canal) or absent (1) (Nesbitt, 2011: 357) (Fig. 45).

See comments in Nesbitt (2011: character 357).

539. Astragalus, posterior groove: present (0); absent (1) (Sereno, 1991; Gower, 1996; Ezcurra, Lecuona & Martinelli, 2010: 138; Nesbitt, 2011: 363) (Fig. 46 of Nesbitt, 2011).

See comments in Nesbitt (2011: character 363).

540. Astragalus, anteromedial corner in proximal view: obtuse (0); acute (1) (Bonaparte, 1976; Sereno, 1991; Juul, 1994; Ezcurra, Lecuona & Martinelli, 2010: 139; Nesbitt, 2011: 361) (Fig. 45).

See comments in Nesbitt (2011: character 361).

541. Astragalus, dorsolateral margin: overlaps the anterior and posterior portions of the calcaneum equally (0); posterior corner dorsally overlaps the calcaneum much more than the anterior portion (1) (Nesbitt et al., 2009; Ezcurra, Lecuona & Martinelli, 2010: 161; Nesbitt, 2011: 360) (Fig. 46 of Nesbitt, 2011).

See comments in Nesbitt (2011: character 360). However, the presence of character-state 1 is found here to be restricted to Riojasuchus tenuisceps among the sampled taxa, contrasting with the broader phylogenetic distribution of the condition described by Nesbitt (2011).

542. Astragalus, articulation with distal tarsal 4: poorly defined (0); well defined (1) (DeBraga & Rieppel, 1997; Müller, 2004: 150; modified from Ezcurra, Scheyer & Butler, 2014: 213 Pritchard et al., 2015: 191).

The region for reception of the distal tarsal 4 on the ventral surface of the astragalus is developed as a ventrally open notch in dorsal/anterior or ventral/posterior view, which defines a transversely concave articular facet, in lepidosauromorphs (e.g., Planocephalosaurus robinsonae: Fraser & Walkden, 1984; Gephyrosaurus bridensis: Evans, 1981) and the tanystropheid Tanystropheus longobardicus (Nosotti, 2007). In other diapsids, the facet for articulation with the distal tarsal 4 is poorly defined on a convex or flat surface on the astragalus.

543. Calcaneum, articular facet for the astragalus: lies completely medial to the fibular facet (0); lies partially ventral to the fibular facet (1) (Parrish, 1993; Nesbitt, 2011: 358) (Fig. 45 of Nesbitt, 2011).

See comments in Nesbitt (2011: character 358).

544. Calcaneum, development of lateral margin: calcaneum terminating in unthickened margin (0); roughened tuberosity present laterally (1) (Pritchard et al., 2015: 188) (Fig. 45).

This character describes the presence of a thickened lateral margin of the astragalus by a roughened tuberosity, which is present—among the taxa sampled here—in Tanystropheus longobardicus (Pritchard et al., 2015), allokotosaurians (e.g., Pamelaria dolichotrachela: ISI R316; Azendohsaurus madagaskarensis: Nesbitt et al., 2015), rhynchosaurs (e.g., Mesosuchus browni: SAM-PK-7416; Howesia browni: SAM-PK-5886), Prolacerta broomi (BP/1/2676), Boreopricea fuenerea (PIN 3708/1), and non-avemetatarsalian archosauriforms (e.g., Proterosuchus alexander: NMQR 1484; Erythrosuchus africanus: BP/1/2096; Tropidosuchus romeri: PVL 4601; Parasuchus hislopi ISI R42; Riojasuchus tenuisceps: PVL 3827) and Marasuchus lilloensis (PVL 3871).

545. Calcaneum, calcaneal tuber (=expansion of the lateral margin of the bone): absent or incipient (0); prominent (1) (Gauthier, 1986; Laurin, 1991: F9; Sereno, 1991; Juul, 1994; Müller, 2004: 75; Reisz, Laurin & Marjanović, 2010: 109; Ezcurra, Lecuona & Martinelli, 2010: 142; Nesbitt, 2011: 373; Ezcurra, Scheyer & Butler, 2014: 108; Pritchard et al., 189) (Figs. 45 and 46).

Figure 46 Right posterior autopodia of Triassic archosauromorphs in (A) posterior/ventral and (B) anterior/dorsal views.

(A) Macrocnemus bassanii (PIMUZ T2472); and (B) Tropidosuchus romeri (PVL 4601). Numbers indicate character-states scored in the data matrix. Scale bars equal 5 mm.

See comments in Nesbitt (2011: character 373).

546. Calcaneum, orientation of calcaneal tuber: lateral, between 0°–35°(0); posterolateral, deflected between 36°–70°(1); posterior, between 71°–90°(2) (Sereno, 1991; Ezcurra, Lecuona & Martinelli, 2010: 70, in part; Nesbitt, 2011: 377; modified from Dilkes & Arcucci, 2012: 83, and Nesbitt et al., 2015: 231), ORDERED. This character is inapplicable in taxa that lack or have an incipient calcaneal tuber (Fig. 45).

See comments in Nesbitt (2011: character 377).

547. Calcaneum, proportions of calcaneal tuber at the midshaft: taller than broad (0); about the same or broader than tall (1); just less than twice the transverse width of the fibular facet (2) (Sereno, 1991; Ezcurra, Lecuona & Martinelli, 2010: 74, in part; Nesbitt, 2011: 376; Dilkes & Arcucci, 2012: 87, in part; Nesbitt et al., 2015: 229, in part). This character is inapplicable in taxa that lack or have an incipient calcaneal tuber (Fig. 45).

See comments in Nesbitt (2011: character 376).

548. Calcaneum, calcaneal tuber distal end: rounded and unexpanded (0); flared, dorsally and/or ventrally (1) (Sereno, 1991; reworded from Ezcurra, Lecuona & Martinelli, 2010: 75; Nesbitt, 2011: 374; Dilkes & Arcucci, 2012: 88). This character is inapplicable in taxa that lack or have an incipient calcaneal tuber (Fig. 45).

A dorsally and/or ventrally flared calcaneal tuber is absent in the non-archosauriform archosauromorphs sampled here (e.g., Azendohsaurus madagaskarensis: Nesbitt et al., 2015; Prolacerta broomi: BP/1/2676), but it is present in “Chasmatosaurus” yuani (IVPP V4067), Shansisuchus shansisuchus (Gower, 1996), Euparkeria capensis (UMCZ T692), proterochampsids (e.g., Tropidosuchus romeri: PVL 4601), phytosaurs (e.g., Parasuchus hislopi: ISI R42), and crocodile-line archosaurs (e.g., Nundasuchus songeaensis: Nesbitt et al., 2014; Aetosauroides scagliai: PVL 2073).

549. Calcaneum, calcaneal tuber distal end in proximal or distal view: tapering or squared (0); expanded (1) (New character). This character is inapplicable in taxa that lack or have an incipient calcaneal tuber (Fig. 45).

The calcaneal tuber is anteroposteriorly/transversely expanded in “Chasmatosaurus” yuani (IVPP V4067), the erythrosuchids Erythrosuchus africanus (BP/1/2096) and Shansisuchus shansisuchus (Gower, 1996), phytosaurs (e.g., Parasuchus hislopi: ISI R42; Smilosuchus gregorii: 18313), ornithosuchids (e.g., Ornithosuchus longidens: Walker, 1964; Riojasuchus tenuisceps: PVL 3827), Nundasuchus songeaensis (Nesbitt et al., 2014), and suchians (e.g., Tufanosuchus dabanensis: IVPP V3237; Gracilisuchus stipanicicorum: PVL 4597; Aetosauroides scagliai: PVL 2052; Batrachotomus kupferzellensis: (Gower & Schoch, 2009); Prestosuchus chiniquensis: UFRGS-PV-0152-T). By contrast, the calcaneal tuber tapers distally or is squared in proximal or distal view in non-archosauriform diapsids, several archosauriform clades (e.g., Euparkeria capensis; proterochampsids), and avemetatarsalians.

550. Calcaneum, distal surface of calcaneal tuber with a vertical median depression: absent (0); present (1) (Parrish, 1993; Juul, 1994; Ezcurra, Lecuona & Martinelli, 2010: 143; Nesbitt, 2011: 375). This character is inapplicable in taxa that lack or have an incipient calcaneal tuber (Figs. 44, 45 of Nesbitt, 2011).

See comments in Nesbitt (2011: character 375).

551. Calcaneum, ventral notch between the main body and the calcaneal tuber: absent (0); present (1) (New character). This character is inapplicable in taxa that lack or have an incipient calcaneal tuber (Fig. 45).

The main body of the calcaneum and the calcaneal tuber are contiguous on the ventral surface of the bone, without a distinct topological differentiation between both regions, in the vast majority of diapsids sampled here. However, there is an anteriorly opened notch that delimits medially the calcaneal tuber on the ventral surface of the bone (=calcaneal notch of Gower, 1996) in Pamelaria dolichotrachela (ISI R316), Proterosuchus alexanderi (NMQR 1484), Garjainia prima (Gower, 1996), Euparkeria capensis (UMCZ T692), and Riojasuchus tenuisceps (PVL 3827).

552. Calcaneum, ventral articular surface for distal tarsal 4 and the distal end of the calcaneal tuber: continuous (0); separated by a clear gap (1); separated by a gap with a laterally and medially delimited ventral fossa (2) (Nesbitt, 2011: 371), ORDERED. This character is inapplicable in taxa that lack or have an incipient calcaneal tuber (Fig. 45 of Nesbitt, 2011).

See comments in Nesbitt (2011: character 371).

553. Calcaneum, fibular facet: slightly convex or flat (0); hemicylindrical (1); concave (2) (modified from Novas, 1989; Parrish, 1993; Juul, 1994; modified from Ezcurra, Lecuona & Martinelli, 2010: 72, 140; Nesbitt, 2011: 378; Dilkes & Arcucci, 2012: 85) (Fig. 45).

See comments in Nesbitt (2011: character 378).

554. Calcaneum, articular facets for the fibula and astragalus: connected by a continuous surface (0); separated (1) (Nesbitt, 2011: 372).

See comments in Nesbitt (2011: character 372).

555. Calcaneum, articular surfaces for fibula and distal tarsal 4: separated by a non-articular surface (0); continuous (1) (Sereno, 1991; Ezcurra, Lecuona & Martinelli, 2010: 71; Nesbitt, 2011: 380; Dilkes & Arcucci, 2012: 84; Nesbitt et al., 2015: 230).

See comments in Nesbitt (2011: character 380).

556. Calcaneum, transverse width of the distal articular surface versus transverse width of the astragalus: 0.28–0.33 (0); 0.42–0.48 (1); 0.54–1.22 (2) (modified from Sereno, 1991; Juul, 1994; Ezcurra, Lecuona & Martinelli, 2010: 141), ORDERED (Fig. 45).

This character describes the relative size of the calcaneum with respect to that of the astragalus. The size of the calcaneum is measured here as the transverse width of its distal articular surface, i.e., excluding the calcaneal tuber (if present) in order to allow the character to be comparable with taxa that lack a calcaneal tuber. The size of the astragalus is measured as the total transverse width of the bone. The calcaneum is subequal in transverse width to or represents more than half of the width of the astragalus in non-archosauriform diapsids (with the exception of Pamelaria dolichotrachela: ISI R316), Erythrosuchus africanus (BP/1/2096), phytosaurs (e.g., Parasuchus hislopi: ISI R42; Smilosuchus gregorii: USNM 18313), ornithosuchids (Riojasuchus tenuisceps: PVL 3827), Nundasuchus songeaensis (Nesbitt et al., 2014), and suchians (e.g., Gracilisuchus stipanicicorum: PVL 4597; Aetosauroides scagliai: PVL 2052; Prestosuchus chiniquensis: UFRGS-PV-0152-T). By contrast, the transverse width of the distal articular surface of the calcaneum represents less than half of the width of the astragalus in Pamelaria dolichotrachela, Proterosuchus alexanderi (NMQR 1484), Vancleavea campi (Nesbitt et al., 2009), Euparkeria capensis (UMCZ T692), Tropidosuchus romeri (PVL 4601), and avemetatarsalians (e.g., Marasuchus lilloensis: PVL 3871; Silesaurus opolensis: ZPAL AbIII/361/18; Herrerasaurus ischigualastensis: PVL 2566, PVSJ 373).

557. Distal tarsals, medial pedal centrale: present and does not contact tibia (0); present and contacts tibia (1); absent as a separate ossification (2) (DeBraga & Rieppel, 1997; Dilkes, 1998: 117; Müller, 2004: 151; Ezcurra, Lecuona & Martinelli, 2010: 77, in part; Nesbitt, 2011: 381, in part; Dilkes & Arcucci, 2012: 89, in part; modified from Ezcurra, Scheyer & Butler, 2014: 214; Pritchard et al., 2015: 184, 192), ORDERED (Figs. 45 and 46).

See comments in Nesbitt (2011: character 381).

558. Distal tarsals, distal tarsal 1: present (0); absent (1) (Gauthier, 1984; Dilkes, 1998; Rieppel, Mazin & Tchernov, 1999; Müller, 2004: 76; Ezcurra, Lecuona & Martinelli, 2010: 78, in part; Nesbitt, 2011: 346, in part; Dilkes & Arcucci, 2012: 90, in part; Ezcurra, Scheyer & Butler, 2014: 215; Pritchard et al., 2015: 193).

See comments in Nesbitt (2011: character 346).

559. Distal tarsals, distal tarsal 2: present (0); absent (1) (Gauthier, 1984; Dilkes, 1998; Ezcurra, Lecuona & Martinelli, 2010: 78, in part; Nesbitt, 2011: 346, in part; Dilkes & Arcucci, 2012: 90, in part; Pritchard et al., 2015: 194).

See comments in Nesbitt (2011: character 346).

560. Distal tarsals, distal tarsal 4 transverse width: broader than distal tarsal 3 (0); subequal to distal tarsal 3 (1) (Sereno, 1991; Juul, 1994; Ezcurra, Lecuona & Martinelli, 2010: 144; Nesbitt, 2011: 347; Dilkes & Arcucci, 2012: 91) (Fig. 46, Fig. 42 of Nesbitt, 2011).

See comments in Nesbitt (2011: character 347).

561. Distal tarsals, articular facet for metatarsal V on distal tarsal 4: more than half of the lateral surface of the bone (0); less than half of the lateral surface of the bone (1) (Sereno, 1991, in part; Ezcurra, Lecuona & Martinelli, 2010: 145, in part; Nesbitt, 2011: 348; Dilkes & Arcucci, 2012: 92, in part) (Fig. 42 of Nesbitt, 2011).

See comments in Nesbitt (2011: character 348).

562. Distal tarsals, proximal surface of distal tarsal 4: flat (0); distinct, proximally raised region on the posterior portion (=heel of Sereno & Arcucci, 1994b) (1) (Nesbitt, 2011: 353) (Fig. 42 of Nesbitt, 2011).

See comments in Nesbitt (2011: character 353).

563. Distal tarsals, distal tarsal 5: present (0); absent (1) (Laurin, 1991: E13; Reisz, Laurin & Marjanović, 2010: 112; Ezcurra, Scheyer & Butler, 2014: 111; Pritchard et al., 2015: 195).

The early diapsids Petrolacosaurus kansensis (Reisz, 1981) and Youngina capensis (Broom, 1921; Smith & Evans, 1996) possess a distal tarsal 5, but this bone is absent in all the archosauromorphs scored here.

564. Pes, foot length (articulated fourth metatarsal and digit) versus tibia-fibula length: >1 (0); <1 (1) (DeBraga & Reisz, 1995: 33, 42; Reisz, Laurin & Marjanović, 2010: 106; cf. Ezcurra, Lecuona & Martinelli, 2010: 80; Ezcurra, Scheyer & Butler, 2014: 105) (Fig. 15).

This length of the fourth metatarsal and digit is lower than that of the tibia or fibula in tanystropheids (e.g., Amotosaurus rotfeldensis: SMNS 54783; Macrocnemus bassanii: PIMUZ T4822; Tanystropheus longobardicus: PIMUZ T2817), Prolacerta broomi (BP/1/2676), proterochampsids (e.g., Chanaresuchus bonapartei: PVL 4575; Pseudochampsa ischigualastensis: Trotteyn & Ezcurra, 2014) and Marasuchus lilloensis (PVL 3870, 3871).

565. Metatarsus, configuration: metatarsals diverging from ankle (0); compact, metatarsals I–IV tightly bunched (1) (Gauthier, 1986; Benton, 2004; Ezcurra, Lecuona & Martinelli, 2010: 79; Nesbitt, 2011: 382; Dilkes & Arcucci, 2012: 93) (Fig. 46).

See comments in Nesbitt (2011: character 382).

566. Metatarsus, metatarsals overlapping proximally: absent (0); present (1) (DeBraga & Reisz, 1995: 43; Reisz, Laurin & Marjanović, 2010: 110; Ezcurra, Scheyer & Butler, 2014: 109) (Fig. 46).

The proximal end of the metatarsals overlaps to each other from medial to lateral in anterior/dorsal or posterior/ventral view in the vast majority of diapsids. However, the metatarsals do not overlap to each other proximally and the contacts between them are mainly anteroposteriorly/dorsoventrally oriented in proximal view in Silesaurus opolensis (ZPAL AbIII/363) and Heterodontosaurus tucki (SAM-PK-K1332).

567. Metatarsus, length of the longest metatarsal versus length of the tibia: 0.20–0.23 (0); 0.29–0.32 (1); 0.37–0.59 (2); 0.62–0.65 (3) (Sereno, 1991; modified from Nesbitt, 2011: 383; modified from Dilkes & Arcucci, 2012: 94), ORDERED (Fig. 15).

See comments in Nesbitt (2011: character 383).

568. Metatarsus, metatarsals I and V mid-shaft diameters: subequal or greater (0); lower (1) than those of metatarsals II–IV (Sereno, 1991; Juul, 1994; Ezcurra, Lecuona & Martinelli, 2010: 146,162; Nesbitt, 2011: 384) (Figs. 47 and 48 of Nesbitt, 2011).

See comments in Nesbitt (2011: character 384).

569. Metatarsus, length of metatarsal I versus metatarsal III: 0.17–0.21 (0); 0.27–0.33 (1); 0.38–0.42 (2); 0.46–0.79 (3); 0.93–0.97 (4) (modified from Sereno, 1991; Dilkes, 1998: 123; Müller, 2004: 173; Modesto & Sues, 2004: 123; cf. Ezcurra, Lecuona & Martinelli, 2010: 147; modified from Nesbitt, 2011: 387; modified from Ezcurra, Scheyer & Butler, 2014: 216), ORDERED (Fig. 46).

See comments in Nesbitt (2011: character 387).

570. Metatarsus, anteromedial portion of the shaft of metatarsal I: smooth or slight ridge (0); distinct, rugose ridge present (1) (Nesbitt, 2011: 386) (Fig. 47 of Nesbitt, 2011).

See comments in Nesbitt (2011: character 386).

571. Metatarsus, length of the metatarsal II versus length of the metatarsal IV: 0.52–0.56 (0); 0.60–0.85 (1); 0.90–1.02 (2); 1.06–1.15 (3) (Gauthier, 1986; modified from Nesbitt, 2011: 395), ORDERED (Fig. 46).

See comments in Nesbitt (2011: character 395).

572. Metatarsus, metatarsal II midshaft diameter: less than or equal to the midshaft diameter of the metatarsals I–IV (0); more than the midshaft diameter of metatarsal I (1) (Nesbitt et al., 2009; Nesbitt, 2011: 388; Dilkes & Arcucci, 2012: 95) (Fig. 46).

See comments in Nesbitt (2011: character 388).

573. Metatarsus, metatarsal IV mid-shaft diameter: subequal to (0); lower than (1) that of metatarsal III (Nesbitt et al., 2009; Ezcurra, Lecuona & Martinelli, 2010: 163; Nesbitt, 2011: 394; Dilkes & Arcucci, 2012: 96) (Fig. 46).

See comments in Nesbitt (2011: character 394).

574. Metatarsus, length of metatarsal IV versus length of metatarsal III: 0.85–1.00 (0); 1.04–1.08 (1); 1.11–1.28 (2); 1.31–1.34 (3) (Laurin, 1991: F10; Reisz, Laurin & Marjanović, 2010: 111; Ezcurra, Lecuona & Martinelli, 2010: 164; modified from Nesbitt, 2011: 393, Dilkes & Arcucci, 2012: 97, Ezcurra, Scheyer & Butler, 2014: 110, and Nesbitt et al., 2015: 232), ORDERED (Fig. 46).

See comments in Nesbitt (2011: character 393).

575. Metatarsus, distal articular surface of metatarsal IV: broader than deep (nearly symmetrical) (0); as broad as deep as or deeper than broad (asymmetrical) (1) (Sereno, 1999; Nesbitt, 2011: 391).

See comments in Nesbitt (2011: character 391).

576. Metatarsus, dorsal prominence separated from the proximal surface by a concave gap in metatarsal V: absent (0); present (1) (Nesbitt, 2011: 397) (Fig. 46, Fig. 47 of Nesbitt, 2011).

See comments in Nesbitt (2011: character 397).

577. Metatarsus, metatarsal V with a hook-shaped proximal end: absent, articular face for distal tarsal 4 aligned to the medial margin of the shaft (0); present, with a gradually medially curved proximal process (1); present, with an abruptly medially flexed proximal process and, as a result, the metatarsal acquires a L-shape in dorsal or ventral view (2) (Laurin, 1991: E14; Sereno, 1991; Juul, 1994; Müller, 2004: 80; Reisz, Laurin & Marjanović, 2010: 113; Ezcurra, Lecuona & Martinelli, 2010: 148; Nesbitt, 2011: 398; Dilkes & Arcucci, 2012: 98; modified from Ezcurra, Scheyer & Butler, 2014: 112; Pritchard et al., 2015: 197, in part) (Fig. 46).

See comments in Nesbitt (2011: character 398). Taxa with a hooked metatarsal V can be distinguished as having two different states. The first state refers to a gradual transition between the medial component of the proximal end of the bone and its shaft, resulting in a rather homogeneously convex lateral margin in dorsal or ventral view. This condition occurs in Gephyrosaurus bridensis (Evans, 1981), Protorosaurus speneri (Gottmann-Quesada & Sander, 2009), Rhynchosaurus articeps (Benton, 1990), Euparkeria capensis (SAM-PK-K8309), and Parasuchus hislopi (ISI R42). By contrast, the proximal end of the metatarsal V is abruptly medially offset from the shaft, resulting in squared proximolateral corner of the bone in dorsal or ventral view, in Macrocnemus bassanii (Rieppel, 1989a), allokotosaurians (e.g., Pamelaria dolichotrachela: ISI R316; Azendohsaurus madagaskarensis: Nesbitt et al., 2015), early rhynchosaurs (e.g., Noteosuchus colletti: AM 3591), Boreopricea funerea (PIN 3708/1), Prolacerta broomi(Gow, 1975), proterosuchids (e.g., Proterosuchus fergusi: SAM-PK-K140), Tasmaniosaurus triassicus (Ezcurra, 2014), erythrosuchids (Gower, 1996), and Batrachotomus kupferzellensis (Gower & Schoch, 2009).

578. Metatarsus, metatarsal V outer process on the proximal lateral margin: absent, smooth curved margin (0); present, prominent pointed process (1) (Pritchard et al., 2015: 196).

The proximal end of the metatarsal V is laterally expanded, bearing a prominent outer process, in non-archosauromorph diapsids (e.g., Petrolacosaurus kansensis: Reisz, 1981; Youngina capensis: Smith & Evans, 1996; Planocephalosaurus robinsonae: Fraser & Walkden, 1984; Gephyrosaurus bridensis: Evans, 1981), Protorosaurus speneri (SMNS 55387, cast of Simon/Bartholomäus specimen), the tanystropheids Amotosaurus rotfeldensis (SMNS 54810) and Tanystropheus longobardicus (Nosotti, 2007), erythrosuchids (Erythrosuchus africanus: BP/1/2096; Shansisuchus shansisuchus: Young, 1964), Vancleavea campi (Nesbitt et al., 2009), Jaxtasuchus salomoni (SMNS 91352C), and Dimorphodon macronyx (NHMUK PV R41212-13). By contrast, the lateral margin of the proximal end of the metatarsal V is straight in the other species sampled here.

579. Metatarsus, metatarsal V lateral plantar tubercle in mature individuals: absent (0); present (1) (Ezcurra, Scheyer & Butler, 2014: 217) (Fig. 46).

The lateral plantar tubercle is a raised, mound-like process that it is placed on the ventral/posterior surface of the metatarsal V. This tubercle is adjacent to the lateral margin of the bone and situated approximately at its mid-length. The lateral plantar tubercle is present in early lepidosauromorphs (Planocephalosaurus robinsonae: Fraser & Walkden, 1984; Gephyrosaurus bridensis: Evans, 1981), tanystropheids (e.g., Macrocnemus bassanii: PIMUZ T2472; Tanystropheus longobardicus: Nosotti, 2007), Azendohsaurus madagaskarensis (Nesbitt et al., 2015), Noteosuchus colletti (Carroll, 1976), Boreopricea funerea (PIN 3708/1) and Erythrosuchus africanus (BP/1/2096).

580. Metatarsus, metatarsal V medial plantar tubercle in mature individuals: absent (0); present (1) (DeBraga & Rieppel, 1997; Müller, 2004: 172; Ezcurra, Scheyer & Butler, 2014: 218) (Fig. 46).

The medial plantar tubercle is a raised, mound-like process adjacent to the medial margin of the ventral/posterior surface of the metatarsal V. This process is usually placed in the transition between the medial process and the shaft in the hook-shaped fifth metatarsals of early lepidosauromorphs (Planocephalosaurus robinsonae: Fraser & Walkden, 1984; Gephyrosaurus bridensis: Evans, 1981) and Macrocnemus bassanii (PIMUZ T2472). This tubercle is absent in other diapsids scored here.

581. Pedal digits, length of digit III versus length of digit IV: 0.64–0.77 (0); 0.82–0.83 (1); 0.87–1.44 (2) (Dilkes, 1998: 124; Ezcurra, Lecuona & Martinelli, 2010: 81; modified from Ezcurra, Scheyer & Butler, 2014: 219), ORDERED.

The length of the metatarsal III is subequal to that of the metatarsal IV in Tanystropheus longobardicus (PIMUZ T2817), Azendohsaurus madagaskarensis (Nesbitt et al., 2015), Prolacerta broomi (BP/1/2676), Chanaresuchus bonapartei (Romer, 1972b), Parasuchus hislopi (ISI R42), Lagerpeton chanaresnsis (PVL 4619) and Heterodontosaurus tucki (SAM-PK-K1332). By contrast, the metatarsal III is considerably shorter than the metatarsal IV (ratio <0.85) in non-archosauromorph diapsids (e.g., Petrolacosaurus kansensis: Peabody, 1952; Youngina capensis: Smith & Evans, 1996), Protorosaurus speneri (SMNS 55387, cast of Simon/Bartholomäus specimen; NHMW 1943I4), Amotosaurus rotfeldensis (SMNS 54783b), and Rhynchosaurus articeps (SHYMS 5).

582. Pedal digits, phalanges on pedal digit V: present and “fully” developed first phalanx (0); present and “poorly” developed first phalanx (1); absent (2) (Gauthier, 1984; Juul, 1994; Ezcurra, Lecuona & Martinelli, 2010: 82; Nesbitt, 2011: 399; Dilkes & Arcucci, 2012: 99, in part), ORDERED (Fig. 46, Figs. 47, 48 of Nesbitt, 2011).

See comments in Nesbitt (2011: character 399).

583. Pedal digits, ratio of lengths of pedal digits V and I: 0.30–0.85 (0); 1.37–3.07 (1) (Juul, 1994; modified from Ezcurra, Lecuona & Martinelli, 2010: 83; modified from Dilkes & Arcucci, 2012: 100). This character is inapplicable in taxa lacking digit V.

The fifth pedal digit is longer than or subequal in length to the first digit in most diapsids, but the first digit is longer in Euparkeria capensis (SAM-PK-8309) and Herrerasaurus ischigualastensis (PVSJ 373).

584. Pedal digits, phalanx V-1: subequal to or shorter than other non-ungual phalanges (0); metatarsal-like, considerably longer than other non-ungual phalanges (1) (Fraser & Rieppel, 2006; Pritchard et al., 2015: 199). This character is inapplicable in taxa lacking a fifth digit.

The first phalanx of the fifth pedal digit is considerably longer than the other non-ungual phalanges and resembles in overall shape metatarsals I–IV (Pritchard et al., 2015) in the tanystropheids Amotosaurus rotfeldensis (SMNS 54810; Fraser & Rieppel, 2006) and Tanystropheus longobardicus (PIMUZ T2817; Wild, 1973; Nosotti, 2007). By contrast, this phalanx is not strongly elongated and similar in morphology to other non-ungual phalanges in other diapsids.

585. Pedal digits, distal articular portion of distal pedal phalanges: lateral and medial sides parallel or near parallel (0); lateral and medial sides converging anteriorly (1) (Nesbitt et al., 2015: 234).

See comments in Nesbitt et al. (2015: character 234).

586. Pedal digits, pedal unguals: weakly transversely compressed, rounded and triangular in cross-section (0); dorsolaterally compressed (1); strongly transversely compressed, with a sharp dorsal keel (2) (Sereno, 1991; Nesbitt, 2011: 400) (Fig. 49 of Nesbitt, 2011).

See comments in Nesbitt (2011: character 400).

587. Pedal digits, ventral tubercle in unguals: absent or small (0); well-developed and extended ventral to the articular portion of the ungual (1) (Nesbitt et al., 2015: 233).

See comments in Nesbitt et al. (2015: character 233).

588. Osteoderms, dorsal osteoderms: absent (0); present, one row (1); present, two rows (2); present, more than two rows (3) (Bennett, 1996; modified from Ezcurra, Lecuona & Martinelli, 2010: 84; Nesbitt, 2011: 401 and 402; Dilkes & Arcucci, 2012: 101 and 102; Nesbitt et al., 2015: 236, in part), ORDERED (Fig. 47, Fig. 49 of Nesbitt, 2011).

See comments in Nesbitt (2011: characters 401 and 402).

589. Osteoderms, sculpture on their external surface: absent (0); present (1) (Parrish, 1993; Ezcurra, Lecuona & Martinelli, 2010: 149). This character is inapplicable in taxa that lack osteoderms (Fig. 47).

Figure 47 Dermal armour of Triassic archosauriforms in (A, C, D) external and (B, E) posterior views.

(A) Partial ventral (left) and dorsal (right) armours of Aetosauroides scagliai (MCP 13a-b-PV); (B) pair of paramedian dorsal osteoderms in subarticulation with the distal end of a neural spine of Archeopelta arborensis (CPEZ-239a); (C, E) dorsal osteoderm of Doswellia kaltenbachi (USNM 244214); and (D) paramedian dorsal osteoderm of Euparkeria capensis (UMCZ T692). Numbers indicate character-states scored in the data matrix and the arrows indicate anterior direction. Scale bars equal 2 cm in (A), 5 mm in (B, C, E), and 2 mm in (D).

The external surface of the osteoderms is smooth in Youngina capensis (Gow, 1975), “Chasmatosaurus” yuani (IVPP V4067), Erythrosuchus africanus (NHMUK PV R3592), Vancleavea campi (Nesbitt et al., 2009), Euparkeria capensis (SAM-PK-5867), Koilamasuchus gonzalezdiazi (MACN-Pv 18119), proterochampsids (e.g., Tropidosuchus romeri: PVL 4601, 4604; Gualosuchus reigi: PVL 4576; Chanaresuchus bonapartei: MCZ 4037, PULR 07, PVL 6244; Rhadinosuchus gracilis: BSPG AS XXV 50, 51), ornithosuchids (e.g., Ornithosuchus longidens: Walker, 1964; Riojasuchus tenuisceps: PVL 3827), Nundasuchus songeaensis (Nesbitt et al., 2014), gracilisuchids (e.g., Turfanosuchus dabanensis: IVPP V3237; Gracilisuchus stipanicicorum: MCZ 4118, PULR 08), loricatans (e.g., Batrachotomus kupferzellensis: Gower & Schoch, 2009; Prestosuchus chiniquensis: UFRGS-PV-0152-T, 0156-T), and Lewisuchus admixtus (PULR 01). By contrast, the external surface of the osteoderms is ornamented with ridges and/or large and deep pits in Cuyosuchus huenei (MCNAM PV 2669), Yarasuchus deccanensis (ISI R334), doswelliids (Archeopelta arborensis: CPEZ-239a; Tarjadia ruthae: MCZ 4076, PULR 063; Jaxtasuchus salomoni: SMNS 91002, 91083, 91352; Doswellia kaltenbachi: USNM 244214), phytosaurs (e.g., Parasuchus hislopi: ISI R42; Smilosuchus gregorii: USNM 18313), and aetosaurs (e.g., Aetosauroides scalgiai: MCP 13a-b-PV; PVL 2059, 2073).

590. Osteoderms, coarse and incised ornamentation composed of central regular pits of subequal size and contour on the external surface of the dorsal osteoderms: absent (0); present (1) (Desojo, Ezcurra & Schultz, 2011: 91). This character is inapplicable in taxa that lack dorsal osteoderms or sculpture on the external surface of the osteoderms (Fig. 47).

See comments in Desojo, Ezcurra & Schultz (2011: 859, 860).

591. Osteoderms, dorsal prominence on the external surface of paramedian osteoderms: absent (0); longitudinal keel, extending along all or most of the anteroposterior length of the osteoderm as a transversely compressed flange (1); blunt, anteroposteriorly restricted eminence (2) (modified from Schoch & Sues, 2014: 116). This character is inapplicable in taxa that lack paramedian osteoderms (Fig. 47).

The external surface of the osteoderms possesses a dorsal prominence that is raised well above the rest of the surface of the dermal scute in several archosauriforms. In particular, this dorsal prominence is developed as a longitudinal keel in Koilamasuchus gonzalezdiazi (MACN-Pv 18119), Vancleavea campi (Nesbitt et al., 2009), Euparkeria capensis (SAM-PK-5867), Tropidosuchus romeri (PVL 4601, 4604), Ornithosuchus longidens (Walker, 1964), Nundasuchus songeaensis (Nesbitt et al., 2014), gracilisuchids (e.g., Turfanosuchus dabanensis: IVPP V3237; Gracilisuchus stipanicicorum: MCZ 4118, PULR 08), and loricatans (e.g., Batrachotomus kupferzellensis: Gower & Schoch, 2009; Prestosuchus chiniquensis: UFRGS-PV-0152-T, 0156-T). By contrast, the dorsal eminence is represented by a blunt and anteroposteriorly short raised portion of the osteoderm in the doswelliids Jaxtasuchus salomoni (SMNS 91002, 91083, 91352) and Doswellia kaltenbachi (USNM 244214), the phytosaur Parasuchus hislopi (ISI R42), and the aetosaur Aetosauroides scalgiai (MCP 13a-b-PV; PVL 2059, 2073).

592. Osteodemrs, paramedian osteoderms: thin (0); very thick (1) (Desojo, Ezcurra & Schultz, 2011: 90). This character is inapplicable in taxa that lack paramedian osteoderms (Fig. 47).

The paramedian osteoderms of most diapsids are very thin, plate-like dermal ossifications, but the paramedian osteoderms of Yarasuchus deccanensis (ISI R334), some proterochampsids (Gualosuchus reigi: PVL 4576; Chanaresuchus bonapartei: MCZ 4037, PULR 07, PVL 6244; Rhadinosuchus gracilis: BSPG AS XXV 50, 51; Pseudochampsa ischigualastensis: Trotteyn & Ezcurra, 2014) and the doswelliids Archeopelta arborensis (CPEZ-239a) and Tarjadia ruthae (MCZ 4076, PULR 063) are dorsoventrally very thick bones. The condition of the latter taxa clearly differs from thin osteoderms above the neural spines that are observed in the axial skeleton of most archosauriforms in lateral view.

593. Osteoderms, relation between paramedian dorsal osteoderms and presacral vertebrae: one to one (includes pairs) (0); more than one osteoderm (1) (Gauthier, 1986; Benton & Clark, 1988; Nesbitt, 2011: 410). This character is inapplicable in taxa that lack paramedian osteoderms (Fig. 49 of Nesbitt, 2011).

See comments in Nesbitt (2011: character 410).

594. Osteoderms, dorsal osteoderm alignment dorsal to the dorsal vertebrae: staggered (0); one to one (1) (Nesbitt, 2011: 411). This character is inapplicable in taxa that lack dorsal osteoderms or have a single row of dorsal osteoderms.

See comments in Nesbitt (2011: character 411).

595. Osteoderms, dimensions of presacral dorsal osteoderms: square-shaped, about equal dimensions (0); longer than wide (1); wider than long (2) (Nesbitt, 2011: 407). This character is inapplicable in taxa that lack dorsal osteoderms (Fig. 47).

See comments in Nesbitt (2011: character 407).

596. Osteoderms, unornamented anterior articular lamina in paramedian osteoderms: absent (0); present (1) (Desojo, Ezcurra & Schultz, 2011: 92). This character is inapplicable in taxa that lack paramedian osteoderms (Fig. 47).

See comments in Desojo, Ezcurra & Schultz (2011: 860).

597. Osteoderms, anterior edge of paramedian presacral osteoderms: straight or rounded (0); with a distinct anterior process (=leaf shaped) (1) (Clark, Sues & Berman, 2001; Nesbitt, 2011: 403). This character is inapplicable in taxa that lack paramedian osteoderms (Fig. 47).

See comments in Nesbitt (2011: character 403).

598. Osteoderms, presacral paramedian osteoderms with a distinct longitudinal bend near the lateral edge: absent (0); present (1) (Clark, Sues & Berman, 2001; Nesbitt, 2011: 404). This character is inapplicable in taxa that lack paramedian osteoderms or possess a single row of osteoderms (Fig. 49 of Nesbitt, 2011).

See comments in Nesbitt (2011: character 404).

599. Osteoderms, appendicular osteoderms: absent (0); present (1) (Heckert & Lucas, 1999; Nesbitt, 2011: 405).

See comments in Nesbitt (2011: character 405).

600. Osteoderms, ventral osteoderms: absent (0); present, scattered, not forming a carapace (1); present, forming a carapace (2) (Heckert & Lucas, 1999; Ezcurra, Lecuona & Martinelli, 2010: 85; Nesbitt, 2011: 409, in part; Dilkes & Arcucci, 2012: 103, in part) (Fig. 47).

See comments in Nesbitt (2011: character 409).

Alternative analyses

The data matrix was built with the aim of conducting three alternative analyses:

Analysis 1 (holotypes only): in this analysis only the holotypes are included for species with potentially problematical hypodigms. Chasmatosuchus magnus and its proposed subjective junior synonym “Gamosaurus lozovskii” are included as separate terminals. The following four terminals are excluded from this analysis: Kadimakara australiensis combined (i.e., combined equals entire hipodigm), Chasmatosuchus magnus combined (i.e., Chasmatosuchus magnus + “Gamosaurus lozovskii”), Garjainia madiba combined and Uralosaurus magnus combined. The resultant data matrix includes 98 operational taxonomic units.

Analysis 2 (complete hypodigms): in this analysis the complete hypodigms are included for all species and a single terminal was used that combines both Chasmatosuchus magnus and “Gamosaurus lozovskii” (i.e., accepting the proposed synonymy of these species). Therefore, the following five terminals are excluded from this analysis: Kadimakara australiensis holotype, Chasmatosuchus magnus, “Gamosaurus lozovskii”, Garjainia madiba holotype, and Uralosaurus magnus holotype. The resultant data matrix includes 97 operational taxonomic units.

Analysis 3 (reduced analysis): in this analysis the complete hypodigms are included for all species (i.e., excluding Kadimakara australiensis holotype, Chasmatosuchus magnus, “Gamosaurus lozovskii”, Garjainia madiba holotype, and Uralosaurus magnus holotype). In addition, 16 terminals are pruned a priori because they represent nomina dubia or terminals with a very large amount of missing data (usually represented by a single isolated bone), the inclusion of which led to major polytomies in the results of the previous two analyses. The pruned taxa were: Eorasaurus olsoni, Prolacertoides jimusarensis, Archosaurus rossicus, Proterosuchus fergusi holotype, “Ankistrodon indicus,” “Exilisuchus tubercularis,” “Blomosuchus georgii,” Vonhuenia friedrichi, Chasmatosuchus rossicus, Chasmatosuchus magnus combined (i.e., Chasmatosuchus magnus + “Gamosaurus lozovskii”), “Chasmatosuchus” vjushkovi, Long Reef proterosuchid, Shansisuchus kuyeheensis, “Dongusia colorata,” Uralosaurus magnus combined, and “Chasmatosaurus ultimus.” Although it is usually recommended to not prune terminals a priori, it is decided here to exclude these taxa a priori because the alternative positions that they adopt in the different most parsimonious trees (MPTs) can cause ambiguities or artefacts in character optimizations that may affect the tree topologies. The main aim of analysis 3 is to allow other workers to use this data matrix in future studies easily, without requiring large amounts of computer memory and lengthy tree searches. The resultant data matrix includes 81 operational taxonomic units.

Tree search, strict reduced consensus and branch support and stability evaluation

The data matrices were analysed under equally weighted parsimony using TNT 1.1 (Goloboff, Farris & Nixon, 2008). A total of 1,800,000 trees was set to be retained in memory, which is the maximum number of trees possible that could be saved on the computer used for these analyses. A first search using the algorithms Sectorial Searches, Ratchet (perturbation phase stopped after 20 substitutions), and Tree Fusing (5 rounds) was conducted, performing 1,000 replications in order to find all tree islands. The best tree or trees obtained at the end of the replicates were subjected to a final round of TBR branch swapping. Branches with a maximum possible length of zero among any of the recovered MPTs were collapsed (rule 3 of Swofford & Begle, 1993; Coddington & Scharff, 1994).

As a measure of branch support, decay indices (=Bremer support) were calculated (Bremer, 1988; Bremer, 1994), and as a measure of branch stability, a bootstrap resampling analysis (Felsenstein, 1985) was conducted, performing 10,000 pseudoreplicates. Bremer support was calculated after searching for suboptimal trees and not with the script that accompanies the program. Both absolute and GC (i.e., difference between the frequency whereby the original group and the most frequent contradictory group are recovered in the pseudoreplicates; Goloboff et al., 2003) frequencies are reported. Taxa with high amounts of missing data may reduce support values not as a result of genuinely weak support of the branch but because of ambiguous optimizations generated by unknown character-states. Accordingly, a second round of decay indices was calculated following the a posteriori pruning of 52 terminals (Table 2), in which the percentage of missing data exceeded the mean of the missing data of the entire data matrix (this cutoff was decided arbitrarily as a compromise between retaining a representative subsample of terminals and aiming to reduce the amount of missing data). Finally, suboptimal alternative topologies of relationships between taxa were explored using heuristic tree searches under monophyly or non-monophyly constraints. This procedure aims to evaluate how many additional steps are necessary to obtain these alternative topologies, such as those required to recover the monophyly of “Prolacertiformes,” “Proterosuchia,” or previous taxonomically inclusive conceptions of Proterosuchidae. These topologically constrained searches were conducted using the taxonomic sample of analysis 3.

The option of “pruned trees” in TNT was used to search for topologically unstable terminals among the recovered MPTs. The presence of multiple terminals with high amount of missing data in the data matrix analysed here, specifically in analyses 1 and 2, may result in several polytomies in the strict consensus of the recovered MPTs. As a result, a series of strict reduced consensus trees (SRCTs) were generated from the recovered MPTs. These SRCTs were obtained by pruning a posteriori the terminals that were recovered in alternative positions among the MPTs (i.e., ‘wildcard taxa’), with the aim of resolving iteratively the strict consensus from more to less inclusive nodes. This protocol results in a series of sequentially more resolved SRCTs that allow the phylogenetic positions of the ambiguously placed terminals to be constrained.

Statistical analyses and graphics

The consistency index (CI) of each character was calculated for one of the MPTs recovered in analysis 3. These CIs were splitted into 12 different anatomical groups, namely: dermal cranial bones, palatoquadrate, braincase, lower jaw, marginal dentition, presacral vertebrae and ribs, sacral and caudal vertebrae and ribs, pectoral girdle, forelimb, pelvic girdle, hindlimb and dermal scutes. The normality and multimodality of the distribution of CIs was tested with a Shapiro–Wilk test (function: shapiro.test) and a Hartigans’ dip test (function: dip.test) in R (R Development Core Team, 2013), respectively, for the entire data set and for each of the 12 groups. Differences in the mean of the groups were tested with a Kruskal–Wallis test and a Kruskal–Wallis multiple comparison test in R (functions: kruskal.test, kruskalmc) because of the lack of normality of 11 of the 12 CI populations. The histogram showing the frequencies of the CIs in the entire data set and the violin box-plots showing the distribution, mean and standard deviation of the CI of each group were built in R (functions: hist, ggplot). Means and standard deviation were also calculated in R. The significance coefficient (α) for the statistical analyses was at the 0.05 level.

Results

The tree searches reached the maximum number of trees that could be retained in memory (1,800,000 MPTs) with the best score hit one time out of the 1,000 replications in analyses 1 and 2. The MPTs of analysis 1 have a length of 2,651 steps, with a consistency index (CI) of 0.2972 and a retention index (RI) of 0.6290, and the MPTs of analysis 2 have a length of 2,664 steps, with a CI of 0.2958 and a RI of 0.6264. Phylogenetic analysis 3 recovered 40 MPTs with a length of 2,646 steps each, with a CI of 0.2978, a RI of 0.6300, and the best score hit one time out of the 1,000 replications. The strict consensus tree of analysis 3 is very well resolved in its entire topology (Figs. 48–51). The topology of the strict consensus tree is identical in analyses 1 and 2 (if the holotypes are replaced by the complete hypodigms), being rather well resolved for relationships among taxa usually considered as non-archosauriform diapsids and more crownward species than proterosuchians and Euparkeria capensis. Massive polytomies are present among taxa historically classified as proterosuchids, erythrosuchids and suchian archosaurs (Fig. 52). Beyond these areas of lack of resolution, the topologies of the strict consensus trees of the three analyses are completely congruent. The higher-level phylogenetic relationships recovered in analyses 1 and 2 and the results of the iterative a posteriori pruning of problematic terminals are discussed as follows.

Figure 48 Strict reduced consensus tree recovered from analysis 3 showing the phylogenetic relationships of archosauromorphs after the a posteriori pruning of Kalisuchus rewanensis and Asperoris mnyama.

Eucrocopodans have been merged into a single terminal in this tree.

Regarding non-archosauriform diapsids, the Permian Youngina capensis and Acerosodontosaurus piveteaui are found as successive sister-taxa of saurians, which results in a paraphyletic “Younginiformes” (Bickelmann, Müller & Reisz, 2009) (Figs. 48, 49 and 52). The position of Youngina capensis as sister-taxon of Acerosodontosaurus and saurians differs from the result recovered by Reisz, Modesto & Scott (2011), in which tangasaurids (including Tangasaurus, Acerosodontosaurus and Hovasaurus) are the sister-taxa of younginids, Claudiosaurus germaini, and saurians. The phylogenetic relationships of Youngina capensis and Acerosodontosaurus piveteaui recovered here should be considered tentative because the character and taxonomic sample of this study is not focused on this part of the diapsid tree. Sauria is composed of Lepidosauromorpha and Archosauromorpha (Gauthier, Kluge & Rowe, 1988), and the enigmatic choristoderans are unambiguously recovered as saurians, as was proposed or quantitatively found by some previous authors (e.g., Romer, 1956; Erickson, 1987; Gauthier, Kluge & Rowe, 1988; Evans, 1990; Müller, 2004). Alternatively, Dilkes (1998) and Gottmann-Quesada & Sander (2009) recovered choristoderans as the sister-taxon of Sauria; three additional steps are necessary to force this position in the current analysis. Within Sauria, choristoderans are alternatively found here as the most basal lepidosauromorphs (Müller, 2004) or the most basal archosauromorphs (Gauthier, Kluge & Rowe, 1988; Neenan, Klein & Scheyer, 2013). The Early Triassic Paliguana whitei and the early rhynchocephalians Planocephalosaurus robinsonae and Gephyrosaurus bridensis are recovered within a monophyletic Lepidosauromorpha, with the latter two species as sister-taxa.

Figure 49 Branch support values shown on the strict reduced consensus tree recovered from analysis 3 after the a posteriori pruning of Kalisuchus rewanensis and Asperoris mnyama.

Numbers above internal branches are Bremer support values above 1 and numbers below internal branches are absolute (left) and GC (right) bootstrap frequencies. Eucrocopodans have been merged into a single terminal in this tree.

The late Permian Aenigmastropheus parringtoni and Protorosaurus speneri are found as successive sister-taxa to all other archosauromorphs. The enigmatic Middle–Late Triassic herbivorous species Pamelaria dolichotrachela, Trilophosaurus buettneri, and Azendohsaurus madagaskarensis are found as more closely related to each other than to any other archosauromorph and support the recent erection of the clade Allokotosauria by Nesbitt et al. (2015). Allokotosauria,Tanystropheidae, Jesairosaurus lehmani and Prolacertoides jimusarensis and more crownward archosauromorphs form a polytomy. Rhynchosauria, Boreopricea funerea and Prolacertidae are successive sister-taxa of Archosauriformes. Within Archosauriformes, there is a massive polytomy at the base of the clade, but Erythrosuchidae, Proterochampsia, and Archosauria are recovered in the strict consensus trees of these two analyses (Fig. 52). All the taxa usually identified as erythrosuchids (with the exception of “Dongusia colorata”, which was more recently reinterpreted as an archosaur; Charig & Reig, 1970; Gower & Sennikov, 2000; Ezcurra, Butler & Gower, 2013) are recovered within a monophyletic Erythrosuchidae, including both species of Garjainia, both species of Shansisuchus, Uralosaurus magnus, Chalishevia cothurnata, Erythrosuchus africanus, Guchengosuchus shiguaiensis, and the putative juvenile specimen of Erythrosuchus africanus (GHG 7433MI). Erythrosuchidae is the sister-taxon of a polytomy composed of Dongusuchus efremovi, Yarasuchus deccanensis, Dorosuchus neoetus, Asperoris mnyama, Euparkeria capensis, and more crownward archosauriforms.

A clade (here referred to as Proterochampsia) formed by the supposedly semi-aquatic doswelliids and proterochampids is found as the sister-taxon of Archosauria (i.e., Ornithodira + Pseudosuchia) (Figs. 50 and 51). The bizarre Late Triassic Vancleavea campi is recovered within this clade of semi-aquatic archosauriforms and as the sister-taxon of all other doswelliids. Phytosaurs are found within Archosauria as the sister-taxon of all other pseudosuchians. The recently described Nundasuchus songeaensis is found as the sister-taxon to the clade formed by ornithosuchids + suchians. By contrast, the first analysis that included this species in a quantitative phylogeny found it as an early suchian (Nesbitt et al., 2014). Several taxa previously identified as proterosuchians—“Chasmatosaurus ultimus” from the Middle Triassic of China, “Dongusia colorata” from the Middle Triassic of Russia, Youngosuchus sinensis from the Middle Triassic of China, and Koilamasuchus gonzalezdiazi from the Middle–Late Triassic of Argentina—are found within Suchia.

Figure 50 Strict reduced consensus tree recovered from analysis 3 showing the phylogenetic relationships of eucrocopodan archosauromorphs after the a posteriori pruning of Kalisuchus rewanensis and Asperoris mnyama.

Non-eucrocopodan diapsids are not skown in this tree.

The “pruned trees” option found 19 and 18 terminals that are responsible for the lack of resolution among non-archosauriform archosauromorphs, basal archosauriforms, and suchians in analyses 1 and 2, respectively. In all cases these terminals possess a large amount of missing data (>84%) (Table 2) and several of them are based only on isolated bones (e.g., Archosaurus rossicus, “Ankistrodon indicus,” “Blomosuchus georgii,” Vonhuenia friedrichi, “Dongusia colorata”). The SRCTs generated by the series of a posteriori prunings show the same topology in analyses 1 and 2 and are discussed together.

Figure 51 Branch support values shown on the strict reduced consensus tree recovered from analysis 3 after the a posteriori pruning of Kalisuchus rewanensis and Asperoris mnyama.

Numbers above internal branches are Bremer supports above 1 and numbers below internal branches are absolute (left) and GC (right) bootstrap frequencies. Non-eucrocopodan diapsids are not skown in this tree.

Figure 52 Strict consensus tree recovered from analyses 1 and 2.

Numbers above internal branches are Bremer supports above 1 and numbers below internal branches are absolute (left) and GC (right) bootstrap frequencies in analysis 2. Erythrosuchids are highlighted in the brown box. Some clades have been condensed into suprageneric terminals to simplify the figure.

The first SRCT was generated after the a posteriori pruning of Prolacertoides jimusarensis, the holotype of Proterosuchus fergusi (SAM-PK-591), “Ankistrodon indicus,” “Blomosuchus georgii” and “Dongusia colorata.” In this SRCT, the higher-level interrelationships among non-archosauriform archosauromorphs are resolved and Proterosuchidae and Paracrocodylomorpha are recovered. The Early Triassic putative proterosuchid Tasmaniosaurus triassicus is found as the sister-taxon of Archosauriformes. Proterosuchidae is unambiguously restricted to the latest Permian–earliest Triassic Archosaurus rossicus, Proterosuchus fergusi, Proterosuchus alexanderi, Proterosuchus goweri and “Chasmatosaurus” yuani. “Chasmatosaurus ultimus” and Koilamasuchus gonzalezdiazi are placed within an unresolved clade nested within Suchia that is also composed of the loricatans Prestosuchus chiniquensis and Batrachotomus kupferzellensis (Fig. 53). Youngosuchus sinensis is found as more closely related to Batrachotomus kupferzellensis than to other species sampled here, bolstering previous claims that the former species was a rauisuchian archosaur rather than an erythrosuchid (Kalandadze & Sennikov, 1985; Gower & Sennikov, 2000).

Figure 53 First strict reduced consensus tree recovered from analyses 1 and 2.

Putative prolacertiforms and proterosuchids are highlighted in the red and blue boxes, respectively. Some clades have been condensed into suprageneric terminals to simplify the figure and their internal relationships are the same as in previous figures.

The second SRCT was generated after the a posteriori pruning of the taxa excluded in the first SRC and Eorasaurus olsoni, “Chasmatosaurus ultimus,” Vonhuenia friedrichi, Chasmatosuchus rossicus, Chasmatosuchus magnus (Chasmatosuchus magnus and “Gamosaurus lozovskii” in the case of analysis 1) and “Chasmatosuchus” vjushkovi (Fig. 54). In this SRCT, three taxa previously identified as proterosuchids—Sarmatosuchus otschevi, Fugusuchus hejiapanensis and Kalisuchus rewanensis—are found as more crownward than proterosuchids sensu stricto and in a polytomy with Cuyosuchus huenei and the clade formed by Erythrosuchidae and more crownward archosauriforms. The aetosaur Aetosauroides scagliai and Koilamasuchus gonzalezdiazi are recovered as sister-taxa of gracilisuchids and paracrocodylomorphs, but the interrelationships between the former two species and other suchians remain unresolved.

Figure 54 Second strict reduced consensus tree recovered from analyses 1 and 2.

Some clades have been condensed into suprageneric terminals to simplify the figure and their internal relationships are the same as in previous figures.

The third SRCT was generated after the a posteriori pruning of the taxa excluded in the second SRC and “Exilisuchus tubercularis,” Archosaurus rossicus, Uralosaurus magnus, and SAM P41754 (the “Long Reef proterosuchid”) (Fig. 55). The topology of this SRCT is identical to that of the SCT of analysis 3, with the only exception of an unresolved clade within Erythrosuchidae. The phylogenetic relationships within the clade that includes tanystropheids are fully resolved, in which Jesairosaurus lehmani and Macrocnemus bassanii are successive sister-taxa of Tanystropheus longobardicus and Amotosaurus rotfeldensis. Within Proterosuchidae, the South African Proterosuchus fergusi, Proterosuchus alexanderi, and Proterosuchus goweri are successive sister-species of the Chinese “Chasmatosaurus” yuani. Within Erythrosuchidae, Guchengosuchus shiguaiensis is recovered as the most basal member of the group and a clade composed of both species of Garjainia and the possible juvenile erythrosuchid from South Africa (GHG 7433MI) is the sister-taxon of the remaining erythrosuchids. Based on this result, GHG 7433MI is here reinterpreted tentatively as Garjainia sp. and a detailed anatomical and taxonomic analysis of the specimen should test this hypothesis in the future. Erythrosuchus africanus, Chalishevia cothurnata, Shansisuchus kuyeheensis and Shansisuchus shansisuchus are grouped in an unresolved clade. Therefore, the erythrosuchid interrelationships found here favour the hypothesis that Erythrosuchus africanus is more closely related to Shansisuchus shansisuchus (Gower & Sennikov, 1996) rather than to Garjainia prima (contra Ezcurra, Lecuona & Martinelli, 2010).

Figure 55 Third strict reduced consensus tree recovered from analyses 1 and 2.

Some clades have been condensed into suprageneric terminals to simplify the figure and their internal relationships are the same as in previous figures.

The fourth SRCT was generated after the a posteriori pruning of the taxa excluded in the third SRC and Kalisuchus rewanensis, Shansisuchus kuyeheensis, and Asperoris mnyama. The aim of this SRC is to resolve as much as possible the tree and have the same taxonomic content of the SRCT of analysis 3, in which Kalisuchus rewanensis and Asperoris mnyama were pruned a posteriori. The topology of the SRCT is almost identical to that of the SRCT of analysis 3, but Fugusuchus hejiapanensis, Sarmatosuchus otschevi, and Cuyosuchus huenei remain in an unresolved position between each other, whereas in the analysis 3 Fugusuchus hejiapanensis and Sarmatosuchus otschevi are recovered as successive sister-taxa of Cuyosuchus huenei and more crownward archosauriforms (Fig. 56). Within Erythrosuchidae, Erythrosuchus africanus is the sister-taxon of Shansisuchus shansisuchus and Chalishevia cothurnata. Dorosuchus neoetus is the sister-taxon of a polytomy composed of Euparkeria capensis, a clade formed by Yarasuchus deccanensis and Dongusuchus efremovi, and more crownward archosauriforms.

Figure 56 Fourth strict reduced consensus tree recovered from analyses 1 and 2.

Some clades have been condensed into suprageneric terminals to simplify the figure and their internal relationships are the same as in previous figures.

The SRCT generated from the 40 MPTs recovered in analysis 3 shows an almost identical and completely consistent topology to that of the fourth SRCTs of analyses 1 and 2 (Figs. 48, 50 and 56). As a result, the a priori pruning of the terminals in analysis 3 makes sense to allow future researchers to use this data matrix in reasonable computer times and using relatively low computer requirements. The diagnoses and unambiguous synapomorphies common to all the MPTs reported as follows are based on the SRCT of analysis 3.

Unnamed clade (Youngina capensis+ Archosauria)

Temporal range. Latest middle–late Permian (Capitanian–Changhsingian, Youngina capensis; Rubidge et al., 2013) to Recent (Passer domesticus).

Synapomorphies. Tooth-bearing portion of the dentary ventrally curved or deflected (267:0→2); dorsal vertebrae with a subcentral foramen on the lateral surface of the centra (355:0→1); ilium with an angle between the anterior margin of the pubic peduncle and the horizontal plane of the pelvic girdle lower than 45°(467:1→0); and ilium without a posteriorly projected heel on the posterior margin of the ischiadic peduncle in lateral view (468:1→0).

Sauria Gauthier, 1984

Definition. The most recent common ancestor of Lepidosauria and Archosauria and all of its descendants (Gauthier, Kluge & Rowe, 1988).

Temporal range. Middle late Permian (middle Wuchiapingian, Protorosaurus speneri; Ezcurra, Scheyer & Butler, 2014) to Recent (Passer domesticus).

Synapomorphies. Anterior process of the quadratojugal absent (146:2→0); extension of the parietal over the interorbital region absent or slight (160:1→0); quadrate shallowly emarginated (176:0→1); and absence of cleithrum (404:0→1).

Lepidosauromorpha Gauthier, 1984

Definition. Sphenodon and squamates and all saurians sharing a more recent common ancestor with them than they do with crocodiles and birds (Gauthier, Kluge & Rowe, 1988).

Temporal range. Early Triassic (Induan–early Olenekian, Paliguana whitei; Damiani et al., 2000; Rubidge, 2005; Lucas, 2010) to Recent (Varanus niloticus).

Synapomorphies. Quadrate with conch (176:1→2); absence of teeth on lateral ramus of the pterygoid (202:1→2); and maxillary tooth crowns with convex distal edge in labial view on at least some anterior crowns (303:0→2).

Synapomorphies of trees that include choristoderans within the clade. Postfrontal reduced to approximately less than half the dimensions of the postorbital (122:0→1); pterygoid anterior ramus (=palatal process) forms oblique suture with palatine but the ramus ends before reaching anterior limit of palatine (193:0→1); contact between the ectopterygoid and the maxilla (206:0→1); dentary with posterocentral process, in which its margins are not confluent with the dorsal or ventral margins of the lower jaw (273:0→1); narrow lateral exposure of the angular in the lower jaw (290:0→1); and plate-like pubic shaft (476:2→0).

Lepidosauria/Rhynchocephalia (Gephyrosaurus bridensis+Planocephalosaurus robinsonae)

Temporal range. Late Triassic (Rhaetian, Planocephalosaurus robinsonae; Whiteside & Marshall, 2008) to Recent (Sphenodon punctatus).

Synapomorphies. Jugal with anterior extension of the anterior process up to or posterior to the level of mid-length of the orbit (95:0→1); and participation of the postfrontal in the border of the supratemporal fenestra (123:0→1).

Archosauromorpha Huene, 1946

Definition. Protorosaurus and all other saurians that are related more closely to Protorosaurus than to Lepidosauria (Dilkes, 1998).

Temporal range. Middle late Permian (middle Wuchiapingian, Protorosaurus speneri; Ezcurra, Scheyer & Butler, 2014) to Recent (Passer domesticus).

Synapomorphies. At least one or more cervical or anterior dorsal with a parallelogram centrum in lateral view, in which the anterior articular surface is situated higher than the posterior one (313:0→1); posterior cervical and anterior dorsal vertebrae with anterior centrodiapophyseal or paradiapophyseal lamina (315:0→1); cervical and/or anterior dorsal vertebrae with posterior centrodiapophyseal lamina (316:0→1); and posterior cervical and/or anterior dorsal vertebrae with prezygodiapophyseal lamina (317:0→1).

Synapomorphies of trees that include choristoderans within the clade. Ornamentation of the dorsal surface of nasals and/or frontals not composed of ridges radiating from centres of growth (6:1→0); quadratojugal with limited posterior extension of the ventral end, ventral condyles of the quadrate broadly visible in lateral view (156:0→1); parietal with a distinct transverse emargination adjacent to the posterior margin of the bone in late ontogeny (166:0→1); occipital condyle of the basioccipital, with a hemispherical articular surface (229:0→1); dentary with a Meckelian groove dorsoventrally centred on the anterior half of the bone (270:1→0); and humerus without entepicondylar foramen (426:0→1).

Choristodera Cope, 1876

Definition. The most recent common ancestor of Lazarussuchus, Cteniogenys and Champsosaurus and all of its descendants (Dilkes, 1998).

Temporal range. Middle Jurassic (late Bathonian, Cteniogenys sp.; Freeman, 1976; Evans, 1990) to Early Miocene (Lazarussuchus; Hecht, 1992). The enigmatic diapsid Pachystropheus rhaeticus from the latest Triassic of Europe has been referred to the Choristodera and would extend back the temporal range of the group into pre-Jurassic times (Storrs & Gower, 1993; Storrs, Gower & Large, 1996). However, the systematic assignment of this species is considered tentative because of the poor anatomical knowledge for this species and the absence of unambiguous choristoderan synapomorphies (Matsumoto & Evans, 2010). Alternatively, Renesto (2005) proposed that Pachystropheus rhaeticus might be a thalattosaur based on anatomical similarities with Endennasaurus.

Synapomorphies. Strongly dorsoventrally compressed skull with mainly dorsally facing antorbital fenestrae and orbits (3:0→1); external nares confluent with each other (9:0→1); prefrontal contact its counterpart in the median line of the skull roof (107:0→1); postorbital bar composed mostly by the postorbital (125:0→1); pineal foramen absent (164:0→2); neomorph ossification between the pterygoid, quadrate and skull roof (184:0→1); contact between vomer and maxilla (186:0→1); and neurocentral sutures of cervical and dorsal vertebrae remain open in sub-adults and adults (312:0→1).

Unnamed clade (Protorosaurus speneri+ Archosauria)

Temporal range. Middle late Permian (middle Wuchiapingian, Protorosaurus speneri; Ezcurra, Scheyer & Butler, 2014) to Recent (Passer domesticus).

Synapomorphies. Dorsal vertebrae without a subcentral foramen on the lateral surface of the centrum (355:1→0); and trochlea (ulnar condyle) of the humerus considerably laterally displaced from the mid-width of the distal end (429:0→1).

Synapomorphy of some trees. Cervical, dorsal, sacral and caudal vertebrae without a notochordal canal piercing the centrum in adults (310:0→1).

Unnamed clade (Tanystropheidae + Archosauria)

Temporal range. Late Permian (Wuchiapingian, Eorasaurus olsoni, youngest boundary of the biostratigraphic range; Ezcurra, Scheyer & Butler, 2014) to Recent (Passer domesticus).

Synapomorphies. Premaxilla with a well-developed postnarial process that forms most of the border of the external naris or excludes the maxilla from participation in the external naris (36:1→2); posterior cervical and anterior dorsal ribs with long and distinct tuberculum (347:0→1); olecranon process of the ulna absent, not ossified or very low (430:1/2→0); ilium with main axis of the postacetabular process mainly posteriorly oriented in lateral or medial view (464:0→1); femur with partially ossified proximal articular surface, being concave and sometimes with a circular pit (491:0→1); and medial pedal centrale absent as a separate ossification (557:0→1).

Synapomorphy of some trees. Premaxilla with a large main body that forms half or more than half of the snout in front of the posterior border of the external naris (27:0→1).

Unnamed clade (Jesairosaurus lehmani+ Tanystropheidae)

Temporal range. Early Triassic (Induan–Olenekian, Augustaburiania vatagini; Sennikov, 2011) to Late Triassic (Norian, Tanytrachelos ahynis; Fraser, Grimaldi & Olsen, 1996).

Synapomorphies. Premaxilla with five or more tooth positions (42:2→1); parietal extends over interorbital region (160:0→1); pterygoid without teeth on the lateral ramus (202:1→2); interclavicle with a median notch on its anterior margin (407:0→1); and entepicondyle of the humerus moderately large in mature individuals (425:1→0).

Tanystropheidae Gervais, 1858

Definition. The most recent common ancestor of Macrocnemus, Tanystropheus and Langobardisaurus and all of its descendants (node-based) (Dilkes, 1998).

Temporal range. Early Triassic (Induan–Olenekian, Augustaburiania vatagini; Sennikov, 2011) to Late Triassic (Norian, Tanytrachelos ahynis; Fraser, Grimaldi & Olsen, 1996).

Synapomorphies. Maxilla with a distinct ascending process that has a posteriorly concave margin (58:0→1); anterodorsal margin of the maxilla concave at the base of the ascending process (59:0→1); neural spine of the axis strongly dorsoventrally short (328:0→1); neural spine of the axis anterodorsally expanded (329:0→1); length of the fourth and fifth cervical centrum versus the height of their anterior articular surface = 2.92–4.12 (331:0→1); epipophysis present in at least the third to fifth cervical vertebrae (336:0→1); anterior and middle postaxial cervical neural spines with an anterior overhang (343:0→1); length of the centrum versus height of the centrum in posterior dorsals = 1.48–2.04 (352:0→1/2); dorsal vertebrae with fan-shaped neural spine in lateral view (363:0→1); length of the transverse process + rib versus length across zygapophyses in anterior caudal vertebrae = 1.51–1.68 (377:1→2); large fenestra between scapula and coracoid immediately anterior to the glenoid region (388:0→1); caudifemoralis brevis muscle origin on the lateroventral surface of the postacetabular process of the ilium dorsally rimed by a tuberosity, but lacking a brevis fossa (465:0→1); thyroid fenestra between pubis and ischium (471:0→1); ischium with a posterior process that extends from the posterodorsal margin (488:0→1); and diameter of the femoral shaft distally narrowed (509:0→1)

Unnamed clade (Amotosaurus rotfeldensis+Tanystropheus longobardicus)

Temporal range. Early Middle Triassic (Anisian, Amotosaurus rotfeldensis; Fraser & Rieppel, 2006) to latest Middle–earliest Late Triassic (Tanystropheus cf. Tanystropheus longobardicus; Rieppel, Jiang & Fraser, 2010).

Synapomorphies. Orbit with elevated rim restricted to the ascending process of the jugal and sometimes also onto the ventral process of the postorbital (17:0→1); multiple maxillary or dentary tooth crowns with longitudinal labial or lingual striations or grooves (306:0→1); lengthes of the fourth or fifth cervical centra versus the heights of their anterior articular surfaces = 6.09–6.80 (331:1→2); cervical neural spines dorsoventrally depressed at their anteroposterior midpoints, leaving them little more than midline dorsal ridges (344:0→1); length of the longest metacarpal versus length of the longest metatarsal = 0.34–0.39 (446:1→0); medial pedal centrale absent as a separate ossification (557:1→2); metatarsal V without a dorsal prominence separated from the proximal surface by a concave gap (576:1→0); and pedal phalanx V-1 metatarsal-like, being considerably longer than other non-ungual phalanges (584:0→1).

Crocopoda new taxon

Definition. All taxa more closely related to Azendohsaurus madagaskarensis Flynn et al., 2010, Trilophosaurus buettneri Case, 1928, Rhynchosaurus articeps Owen, 1842 and Proterosuchus fergusi Broom, 1903a than to Protorosaurus speneri Meyer, 1830 or Tanystropheus longobardicus Bassani, 1886 (stem-based).

Temporal range. Late Permian (Wuchiapingian, Eorasaurus olsoni, youngest boundary of the biostratigraphic range; Ezcurra, Scheyer & Butler, 2014) to Recent (Passer domesticus).

Diagnosis (syanpomorphies). Dorsal surface of the temporal region of the skull with a supratemporal fossa immediately medial or anterior to the supratemporal fenestra (8:0/2→1); orbit with elevated rim restricted to the ascending process of the jugal and sometimes also onto the ventral process of the postorbital (17:0→1); postnarial process of the premaxilla wide, plate-like (37:1→0); posterolateral process of the parietal dorsoventrally deep, being plate-like in occipital view and subequal to the height of the supraoccipital (168:0→1); paroccipital process of the opisthotic anteroposteriorly-flattened distally (215:0→1); dorsal end of the exoccipital with a dorsomedially inclined process that forms a transversely broad contact with more dorsal occipital elements (219:0→1); anterior inferior process of the prootic well developed (255:0→1); retroarticular process of the hemimandible upturned (284:0→1); ankylothecodont tooth implantation (299:0→1); tibial and fibular facets of the astragalus separated from the anterior hollow by a prominent ridge (535:0→1); astragalus with posterior groove (539:1→0); and calcaneum with a prominent calcaneal tuber (545:0→1).

Remarks. The lack of consensus in the phylogenetic relationships of non-archosauriform archosauromorphs hampered the erection of a taxon framed between Archosauromorpha and Archosauriformes despite of the large number of lineages enclosed in this part of the diapsid tree. Recent phylogenetic analyses and this study recovered three main non-archosauriform archosauromorph clades, namely Tanystropheidae (or its kin), Allokotosauria and Rhynchosauria (Pritchard et al., 2015; Nesbitt et al., 2015). In addition, this study found several less specious or monospecific branches, such as Prolacertidae and those represented by Aenigmastropheus parringtoni, Protorosaurus speneri, Boreopricea funerea and Tasmaniosaurus triassicus. Therefore, it is here considered that the erection of a new higher-level taxon will be useful based on the large number of different lineages currently recognized in this part of the tree. As defined, Crocopoda is a stem-based clade that aims to exclude several species historically grouped within “Protorosauria”/“Prolacertiformes” (e.g., Protorosaurus speneri, tanystropheids) and include allokotosaurians, rhynchosaurs, prolacertids and archosauriforms. Crocopoda means “crocodile foot” and is derived from the Latin word “crocodilus” and the Greek word “pous.” This name refers to the fact that members of this clade possess a prominent calcaneal tuber, which is a plesiomorphic characteristic retained by the crocodile-line of Archosauria. It should be noted that the presence of a calcaneal tuber is a character-state that currently diagnoses Crocopoda, but it does not define it. As a result, the presence of a calcaneal tuber in some non-crocopodan archosauromorphs (e.g., Tanytrachelos ahynis: Pritchard et al., 2015; a preliminary reported and still unnamed new species from the Middle–Late Triassic of Kyrgyzstan: Buchwitz & Ezcurra, 2014) and a morphological analogous structure in extant varanids (Sullivan, 2010) does not invalidate this taxon nor does it affect its taxonomic utility or content.

Allokotosauria Nesbitt et al., 2015

Definition. The least-inclusive clade containing Azendohsaurus madagaskarensis Flynn et al., 2010 and Trilophosaurus buettneri Case, 1928, but not Tanystropheus longobardicus Bassani, 1886, Proterosuchus fergusi Broom, 1903a, Protorosaurus speneri Meyer, 1830, or Rhynchosaurus articeps Owen, 1842 (Nesbitt et al., 2015).

Temporal range. Early Triassic (Olenekian, Coelodontognathus donensis; Arkhangelskii & Sennikov, 2008) to Late Triassic (Trilophosaurus buettneri; Spielmann et al., 2007; Ramezani et al., 2011).

Synapomorphies. Maxilla with a concave anterodorsal margin at the base of the ascending process (59:0→1); maxilla with 10–22 tooth positions (75:3→1/2); jugal with an ascending process that forms the entire anterior border of the infratemporal fenestra (99:0→1); lateral surface of the orbital margin of the prefrontal with a rugose sculpturing (111:0→1); posterodorsal portion of the squamosal with a supratemporal fossa (149:0→1); quadrate with a posteriorly hooked proximal end in lateral view (180:0→1); height and dimetre of the teeth on the palatine, ventral surface of the anterior ramus of the pterygoid and vomer similar to those of the marginal dentition (189:0→1); palatine transversely broad, in which this bone is the main component of the palate posteriorly to the choanae and the anterior ramus of the pterygoid is splint-like (190:0→1); basal tubera of the braincase partially connected to each other (227:0→1); parabasisphenoid oblique, with a posterodorsally-to-anteroventrally oriented main axis (235:0→1); retroarticular process of the hemimandible anteroposteriorly short, being poorly developed posteriorly to the glenoid fossa (283:2→1); distal edge of the maxillary tooth crowns convex in at least some anterior tooth crowns in labial view (303:0→2); posterior cervical and/or anterior dorsal vertebrae with a postzygodiapophyseal lamina (318:0→1); diapophysis and parapophysis of anterior dorsal vertebrae expanded on stalks (356:0→1); coracoid with a subglenoid lip strongly laterally developed as a shelf-like structure, more developed than the supraglenoid lip on the scapula (400:0→1); ulna with a lateral tuber (=radius tuber) on the proximal portion (433:0→1); manual unguals distinctly longer than the last non-ungual phalanx of the same digit (451:0→1); manual unguals trenchant on digits I–III (452:0→1); femoral distal articular surface uneven, lateral (=fibular) condyle projecting distally distinctly beyond medial (=tibial) condyle (512:1→0); distal articular portion of distal pedal phalanges with anteriorly converging lateral and medial sides (585:0→1); and pedal unguals with a well developed ventral tubercle that extends ventral to the articular portion of the ungual (587:0→1).

Unnamed clade (Azendohsaurus madagaskarensis+Trilophosaurus buettneri)

Temporal range. Late Middle Triassic (Azendohsaurus madagaskarensis; Flynn et al., 1999; Flynn et al., 2000) to Late Triassic (Trilophosaurus buettneri; Spielmann et al., 2007; Ramezani et al., 2011).

Synapomorphies. Maxilla with a posterior end of the horizontal process distinctly ventrally deflected from the main axis of the alveolar margin (64:0→1); fusion between opisthotic and exoccipital (=otoccipital) (211:0→1); paroccipital processes of the opisthotics extend laterally forming aproximately a 90°angle with the parasagittal plane (213:1→0); basioccipital and parabasisphenoid tightly sutured, sometimes by an interdigitated suture, or both bones fused with each other in mature individuals (225:0→1); minimum height of the dentary versus length of the alveolar margin (including edentulous anterior end if present) = 0.22–0.29 (266:1→2); posterior surangular foramen on the lateral surface of the bone (289:0→1); atlantal centrum fused to the axial intercentrum in mature individuals (326:0→1); third to fifth cervical vertebrae with epipophysis (336:0→1); anterior and middle postaxial cervical neural spines with an anterior overhang (343:0→1); primordial sacral rib one longer anteroposteriorly than primordial sacral rib two in dorsal view (372:1→0); second sacral rib not bifurcated distally (=single unit) (373:1→0); minimum anteroposterior length of the scapula at the level of the constriction distal to the glenoid less than half the proximodistal length of the bone (391:0→1); ulna with a prominent olecranon process, but lower than its anteroposterior depth at base (430:0→1); femur with a well ossified proximal articular surface, being flat or convex (491:1→0); and femur with a crest-like attachment of the caudofemoralis musculature on the posterior surface of the bone and with intertrochanteric fossa (=internal trochanter), and not convergent with proximal end (504:0→1).

Unnamed clade (Rhynchosauria + Archosauria)

Temporal range. Late Permian (Wuchiapingian, Eorasaurus olsoni, youngest boundary of the biostratigraphic range; Ezcurra, Scheyer & Butler, 2014) to Recent (Passer domesticus).

Synapomorphies. Snout with a subnarial foramen bordered by the maxilla but not the premaxilla (25:0→2); main body of the premaxilla slightly downturned, in which the alveolar margin is angled at approximately 20°to the alveolar margin of the maxilla (29:0→1); supratemporal bone present (157:2→1); interlaced overlap between ectopterygoid and pterygoid (204:0→1); ectopterigoid with a posterior expansion that contacts the jugal (207:0→1); cervical and/or anterior dorsal vertebrae without posterior centrodiapophyseal lamina (316:1→0); and pectoral girdle with a notch on the anterior end of the suture between the scapula and coracoid (385:1→0).

Rhynchosauria Osborn, 1903

Previous definition. The most recent common ancestor of Mesosuchus and Howesia and all of its descendants (node-based) (Dilkes, 1998).

New definition. All taxa more closely related to Rhynchosaurus articeps Owen, 1842 than to Trilophosaurus buettneri Case, 1928, Prolacerta broomi Parrington, 1935 or Crocodylus niloticus Laurenti, 1768 (stem-based). A new definition for Rhynchosauria is here proposed because in the phylogenetic hypothesis recovered here the previous definition would exclude Noteosuchus colletti from the clade, and this species has been considered as a rhynchosaur for more than 100 years (Watson, 1912a; Broom, 1925; Carroll, 1976; Dilkes, 1998; Ezcurra, Scheyer & Butler, 2014).

Temporal range. Earliest Triassic (Indian–early Olenekian, Noteosuchus colletti; Rubidge, 2005) to middle Late Triassic (early Norian, Teyumbaita sulcognathus; Langer et al., 2007; Martínez et al., 2011).

Synapomorphies. Length of the transverse process + rib versus length across zygapophyses in anterior caudal vertebrae = 1.51–1.68 (377:1→2); and length of metatarsal I versus length of metatarsal III = 0.38–0.42 (569:3→2).

Unnamed clade (Mesosuchus browni+ Rhynchosauridae)

Temporal range. Early Middle Triassic (early Anisian, Mesosuchus browni; Rubidge, 2005) to middle Late Trassic (early Norian, Teyumbaita sulcognathus; Langer et al., 2007; Martínez et al., 2011).

Synapomorphies. Neural spine height versus anteroposterior length at its base in anterior caudal vertebrae = 2.36–2.65 (379:0→1); and main axis of the postacetabular process of the ilium posterodorsally oriented in lateral or medial view (464:1→0).

Unnamed clade (Howesia+ Rhynchosauridae)

Temporal range. Early Middle Triassic (early Anisian, Howesia browni; Hancox, 2000) to middle Late Trassic (early Norian, Teyumbaita sulcognathus; Langer et al., 2007; Martínez et al., 2011).

Synapomorphies. Dorsal surface of the temporal region bearing a thin, blade-like median sagittal crest (8:1→2); alveolar margin of the maxilla distinctly convex in lateral view (68:0→1); maxilla with tooth plate (72:0→1); teeth on occlusal and lingual surfaces of the maxilla (74:0→1); anterior margin of the nasals transverse, with little convexity, at midline (78:0→1); and scapula without strong curvature or inflexion between the proximal end and the posterior margin of the scapular blade (389:1→0).

Unnamed clade (Eohyosaurus wolvaardti+ Rhynchosauridae)

Temporal range. Early Middle Triassic (early Anisian, Eohyosaurus wolvaardti; Hancox, 2000) to middle Late Trassic (early Norian, Teyumbaita sulcognathus; Langer et al., 2007; Martínez et al., 2011).

Synapomorphy. Jugal with an ascending process that forms the entire anterior border of the infratemporal fenestra (99:0→1).

Rhynchosauridae Huxley, 1859

Definition. The most recent common ancestor of Rhynchosaurus, Stenaulorhynchus, “Scaphonyx” and Hyperodapedon and all of its descendants (node-based) (Dilkes, 1998).

Temporal range. Early Middle Triassic (Anisian, Rhynchosaurus articeps; Benton, 1990) to middle Late Trassic (early Norian, Teyumbaita sulcognathus; Langer et al., 2007; Martínez et al., 2011).

Synapomorphies. Orbit with well-developed elevated rim along the jugal, postorbital, frontal, prefrontal, and lacrimal (17:1→2); jugal without multiple pits on the lateral surface of its main body (98:1→0); posterior process of the jugal with a distinct lateroventral orientation with respect to the sagittal axis of the snout (101:0→1); medial process of the squamosal long, forming entirely or almost entirely the posterior border of the supratemporal fenestra (140:0→1); parietal with a dorsoventrally low posterolateral process, usually considerably lower than the supraoccipital (168:1→0); teeth absent on the vomer (187:2→3); teeth absent on the palatine and ventral surface of the anterior ramus of the pterygoid (188:0→1); inverted V-shaped supraoccipital in occipital view (208:0→1); distinct dorsal process behind the alveolar margin of the lower jaw formed by a dorsally well-developed surangular (261:0→1); and blade and groove occlusion between dentary and maxillary teeth (280:1→2).

Unnamed clade (Boreopricea funerea+ Archosauria)

Temporal range. Late Permian (Wuchiapingian, Eorasaurus olsoni, youngest boundary of the biostratigraphic range; Ezcurra, Scheyer & Butler, 2014) to Recent (Passer domesticus).

Synapomorphies. Premaxilla with a palatal process on the medial surface (41:0→1); premaxilla with five or more tooth positions (42:2→1); stick-like vomer, being transversely narrower than the choanas (185:0→1); pterygoids remain separate from each other along their entire length (192:0→1); pterygoid with a row of fang-like teeth on the medial edge of the anterior ramus (199:0→1); cervical and dorsal vertebrae with distinct mammillary processes on the lateral surface of the neural spine (320:0→2); humerus with torsion between its proximal and distal ends of approximately 45°or more (415:1→0); and humerus with a moderately large entepicondyle in mature individuals (425:1→0).

Unnamed clade (Prolacertidae + Archosauria)

Temporal range. Late Permian (Wuchiapingian, Eorasaurus olsoni, youngest boundary of the biostratigraphic range; Ezcurra, Scheyer & Butler, 2014) to Recent (Passer domesticus).

Synapomorphies. Dentary with large foramina aligned in two distinct rows starting on the anteroventral corner of the bone (268:0→1); and presence of postaxial cervical intercentra (346:1→0).

Prolacertidae Parrington, 1935

New definition. All taxa more closely related to Prolacerta broomi Parrington, 1935 than to Protorosaurus speneri Meyer, 1832, Tanystropheus longobardicus Bassani, 1886, Proterosuchus fergusi Broom, 1903a or Euparkeria capensis Broom, 1913 (stem-based).

Temporal range. Early Triassic (Prolacerta broomi; Rubidge, 2005).

Synapomorphies. Parietals strongly entering between both frontals, forming an acute-angled suture (116:0→1); and parietal extends over interorbital region (160:0→1).

Comments. This result supports previous claims of very close affinities between Prolacerta broomi and Kadimakara australiensis (Bartholomai, 1979).

Unnamed clade (Tasmaniosaurus triassicus+ Archosauria)

Temporal range. Late Permian (Wuchiapingian, Eorasaurus olsoni, youngest boundary of the biostratigraphic range; Ezcurra, Scheyer & Butler, 2014) to Recent (Passer domesticus).

Synapomorphies. Antorbital fenestra present (13:0→1); maxilla with a distinct ascending process that has a posteriorly concave margin (58:0→1); pineal foramen absent (164:0→2); postparietal present (171:2→0); dentary with a posterocentral processes, in which its margins are not confluent with the dorsal or ventral margin of the lower jaw (273:0→1); serrations on the maxillary and dentary tooth crowns (304:0→1/2); and lateral fossa on the centrum below the neurocentral suture in dorsal vertebrae (354:0→1).

Comments. Tasmaniosaurus triassicus was originally interpreted as a proterosuchid archosauriform, but is recovered here as the sister-taxon of Archosauriformes. As a result, character-states usually considered as synapomorphic of Archosauriformes are optimized here as apomorphic of this more inclusive clade (e.g., antorbital fenestra, serrated teeth; Dilkes, 1998; Nesbitt, 2011).

Archosauriformes Gauthier, Kluge & Rowe, 1988

Definition. The least inclusive clade containing Crocodylus niloticus Laurenti, 1768, and Proterosuchus fergusi Broom, 1903a (node-based) (Nesbitt, 2011).

Temporal range. Latest Permian (Changhsingian, Archosaurus rossicus; Rubidge, 2005; Sennikov & Golubev, 2012) to Recent (Passer domesticus).

Synapomorphies. Interdental plates present (restricted to the anterior end of the dentary in basal forms) (1:0→1); strongly downturned premaxillary main body (29:1→2); orbital border of the frontal absent or anteroposteriorly short in mature individuals (114:1→0); and tooth bearing portion of the dentary distinctly dorsally curved for all or most of its anteroposterior length (267:0→1).

Proterosuchidae Huene, 1908

Definition. All taxa more closely related to Proterosuchus fergusi Broom, 1903a than to Erythrosuchus africanus Broom, 1905, Crocodylus niloticus Laurenti, 1768 or Passer domesticus Linnaeus, 1758 (stem-based) (Ezcurra, Butler & Gower, 2013).

Temporal range. Latest Permian (Changhsingian, Archosaurus rossicus; Rubidge, 2005; Sennikov & Golubev, 2012) to earliest Triassic (Induan–?early Olenekian: Proterosuchus fergusi: Rubidge, 2005).

Synapomorphies. Alveolar margin of the premaxilla does not reach the contact with the maxilla and forms a diastema (=subnarial gap) (26:0→1); base of the prenarial process of the premaxilla anteroposteriorly deep (35:0→1); lateroventrally opening anterior alveoli in the premaxilla in mature individuals (44:0→1); maxilla extends at level with or posterior to posterior orbital border in mature individuals (71:1→0); length of the posterior process of the jugal versus the height of its base = 4.07–5.37 (100:1→2); distal half of the posterior process of the jugal subrectangular (102:0→1); pineal fossa on the median line of the dorsal surface of the parietal (162:0→1); prezygodiapophyseal lamina absent in posterior cervical or anterior dorsal vertebrae (317:1→0); interclavicle with rather sharp angles between lateral and posterior processes (409:0→1); posterior ramus of the interclavicle with little change in width along its entire length (411:1→0); and proximal surface of the femur with transverse groove (495:0→1).

Unnamed clade (Proterosuchus alexanderi+Proterosuchus goweri+ “Chasmatosaurus” yuani)

Temporal range. Earliest Triassic (Induan–?early Olenekian: Proterosuchus alexanderi: Rubidge, 2005).

Synapomorphies. Angle between the posterior margins of the proximal and distal ends of the quadrate = 143–158°(177:2→3); mammillary processes of the neural spine of cervical vertebrae extend posteriorly from the fifth presacral (345:2→1); and mammillary processes of the neural spine of dorsal vertebrae extend up to the thirteenth presacral (365:2→3).

Unnamed clade (Proterosuchus goweri+ “Chasmatosaurus” yuani)

Temporal range. Earliest Triassic (Induan–?early Olenekian: Proterosuchus goweri: Rubidge, 2005).

Synapomorphies. Ventral ramus of the opisthotic rod-like and robust (217: 0→3).

Unnamed clade (Fugusuchus hejiapanensis+ Archosauria)

Temporal range. Late Early Triassic (late Olenekian, Garjainia prima; Gower & Sennikov, 2000) to Recent (Passer domesticus).

Synapomorphies. Dorsal surface of the skull roof without supratemporal fossa immediately medial or anterior to the supratemporal fenestra (8:1→0); maxilla with 15–22 tooth positions (75:3→2); fusion between opisthotic and exoccipital (=otoccipital) (211:0→1); basal tubera of the basioccipital partially connected to each other (227:0→1); and intertuberal plate of the parabasisphenoid straight (237:2→1).

Unnamed clade (Sarmatosuchus otschevi+ Archosauria)

Temporal range. Late Early Triassic (late Olenekian, Garjainia prima; Gower & Sennikov, 2000) to Recent (Passer domesticus).

Synapomorphies. Basioccipital and parabasisphenoid tightly sutured, sometimes by an interdigitated suture, or both bones fused with each other in mature individuals (225:0→1); parabasisphenoid oblique, with a posterodorsally-to-anteroventrally oriented main axis (235:0→1); and parabasisphenoid without parasphenoid crests, so that there is no ventral floor for the vidian canal (246:1→0).

Unnamed clade (Cuyosuchus huenei+ Archosauria)

Temporal range. Late Early Triassic (late Olenekian, Garjainia prima; Gower & Sennikov, 2000) to Recent (Passer domesticus).

Synapomorphies. Cervical and/or anterior dorsal vertebrae with a posterior centrodiapophyseal lamina (316:0→1); and scapula without strong curvature or inflexion between the proximal end and the posterior margin of the scapular blade (389:1→0).

Unnamed clade (Erythrosuchidae + Archosauria)

Temporal range. Late Early Triassic (late Olenekian, Garjainia prima; Gower & Sennikov, 2000) to Recent (Passer domesticus).

Synapomorphies. Second rib without a distal bifurcation (=single unit) (373:1→0); and scapula with a minimum anteroposterior length of the scapular blade less than half the proximodistal length of the bone (391:0→1).

Erythrosuchidae Watson, 1917

Definition. All taxa more closely related to Erythrosuchus africanus Broom, 1905 than to Proterosuchus fergusi Broom, 1903a, or Passer domesticus Linnaeus, 1758 (stem-based) (Ezcurra, Lecuona & Martinelli, 2010).

Temporal range. Late Early Triassic (late Olenekian, Garjainia prima; Gower & Sennikov, 2000) to late Middle Triassic (Ladinian, Chalishevia cothurnata; Gower & Sennikov, 2000).

Synapomorphies. Alveolar margin on the anterior third of the maxilla (anterior to the level of the anterior border of the antorbital fenestra) abruptly upturned (70:0→1); parietal with a strongly transversely convex dorsal margin of the posterolateral process, which is elevated from the median line of the posterior margin of the skull roof (169:0→1); absence of teeth on the palatine and ventral surface of the anterior ramus of the pterygoid (188:0→6); articular with a ventromedially directed process (295:0→1); and distal end of the radius strongly anteroposteriorly expanded (438:0→1).

Synapomorphies present in only some trees. Premaxilla with a postnarial process that fits into slot in the nasal (39:0→2); lateral surface of the anterior and horizontal processes of the maxilla with lateroventrally facing neurovascular foramina that extend ventrally as deep, well-defined grooves (53:0→1).

Unnamed clade (Shansisuchus shansisuchus+Garjainia prima)

Temporal range. Late Early Triassic (late Olenekian, Garjainia prima; Gower & Sennikov, 2000) to late Middle Triassic (Ladinian, Chalishevia cothurnata; Gower & Sennikov, 2000).

Synapomorphies. Maxillo-nasal tuberosity delimiting anteriorly the antorbital fossa (46:0→1); anterior margin of the base of the ascending process of the maxilla sub-vertical (58:1→2); edentulous anterior portion of the ventral margin of the maxilla (69:0→1); ventral surface of the frontal with a median longitudinal canal for the passage of the olfactory tract only slightly constricted and without olfactory bulb impressions and distinct semilunate posteromedially-to-anterolaterally oriented ridges on the orbital roof (120:0→1); pineal fossa on the median line of the dorsal surface of the parietal (162:0→1); lateral surface of the surangular with a laterally projecting shelf that possesses a strongly convex lateral edge (286:2→3); anterior cervical vertebrae with a median longitudinal keel on the ventral surface of the centrum that extends ventral to the centrum rim in at least one vertebra (327:1→2); and postaxial cervical vertebrae with a shallow, posterolaterally facing fossa on the posterior portion of the neural arch ventral to the postzygapophysis (335:0→1).

Garjainia (Garjainia prima+Garjainia madiba+ GHG 7433MI)

Temporal range. Late Early Triassic (late Olenekian, Garjainia prima and Garjainia madiba; Gower & Sennikov, 2000; unknown exact age for GHG 7433MI).

Synapomorphy. Supratemporal fossa immediately medial or anterior to the supratemporal fenestra on the dorsal surface of the skull roof (8:0→1).

Synapomorphies present in only some trees. Premaxilla with a longitudinal groove placed approximately at mid-height and extending along most of the length of the lateral surface of the main body of the bone (31:0→1); posterior portion of the maxilla ventral to the antorbital fenestra expands dorsoventrally towards the distal end of the horizontal process with a straight ventral margin of the antorbital fenestra (63:2→3); quadrate medial condyle distinctly more distally projected than the lateral one (183:0→1); fossa immediately lateral to the foramen magnum on the occipital surface of the opisthotic (216:0→1); ventral ramus of the opisthotic rod-like and very robust (217:4→3); basal tubera of the basioccipital clearly separated from each other (227:1→0); ventral surface of the parabasispheniod with a pair of thick crests running along the ventrolateral border of the basisphenoid body and framing the ventromedial floor of the vidian canal (246:0→1); basipterygoid processes of the parabasisphenoid oriented posterolaterally (248:0→1); distinct longitudinal lamina extending along the lateral surface of the centrum at mid-height in anterior and middle postaxial cervical vertebrae (340:0→1); semicircular preacetabular process of the ilium in lateral view (461:1→0); and prominent tuberosity for the attachment of the ambiens muscle in the pubis of mature individuals (474:1→0).

Unnamed clade (Erythrosuchus africanus+Shansisuchus shansisuchus)

Temporal range. Early Middle Triassic (early Anisian, Erythrosuchus africanus; Rubidge, 2005) to late Middle Triassic (Ladinian, Chalishevia cothurnata; Gower & Sennikov, 2000).

Synapomorphies. Premaxilla with a peg on the posterior edge of the main body (33:0→1); participation of the nasal in the dorsal border of the antorbital fossa (83:0→1); posterior termination of the posterior process of the jugal posterior to the infratemporal fenestra (106:0→1); ventral process of the squamosal forms almost the entire posterior border of the infratemporal fenestra (146:1→2); squamosal with a posterodorsally-to-anteroventrally oriented tuck on the lateral surface of the ventral process (147:0→1); quadratojugal with a depression along the posterior half of the ascending process up to the exposed lateral surface of its distal tip (155:0→1); ventral ramus of the opisthotic covered by the lateralmost edge of the exoccipital in posterior view (218:0→1); distal end of the exoccipital contacts with its counterpart along the entire dorsal surface of the basioccipital, excluding the basioccipital from the floor of the endocranial cavity and the dorsal surface of the occipital condyle (221:1→2); pseudolagenar recess absent in the braincase (223:0→1); base of the cultriform process of the parabasisphenoid tall, with the dorsal edge extending between the clinoid processes and the ventral part of the crista prootica (243:0→1); external foramina for the passage of the abducens nerves (CN VI) open anteriorly (251:1→0); prootic with a ridge on the lateral surface of the inferior anterior process ventral to the trigeminal foramen (256:0→1); dentary with mostly straight tooth bearing portion: (267:1→0); cervico-dorsal transition (=pectoral centra) with a very strong anteroposterior compression of the centra, being considerably anteroposteriorly shorter than tall (311:0→1); and calcaneum with an expanded distal end of the calcaneal tuber in proximal or distal view (549:0→1).

Unnamed clade (Shansisuchus shansisuchus+Chalishevia cothurnata)

Temporal range. Early Middle Triassic (late Anisian, Shansisuchus shansisuchus; Rubidge, 2005; Ezcurra, Butler & Gower, 2013) to late Middle Triassic (Ladinian, Chalishevia cothurnata; Gower & Sennikov, 2000).

Synapomorphies. Secondary antorbital fenestra immediately anterior to the antorbital fenestra (15:0→1); and postaxial cervical vertebrae without excavation immediately lateral to the base of the neural spine (337:1→0).

Eucrocopoda new taxon

Definition. All taxa more closely related to Euparkeria capensis Broom, 1913, Proterochampa barrionuevoi Reig, 1959, Doswellia kaltenbachi Weems, 1980; Parasuchus hislopi Lydekker, 1885, Passer domesticus Linnaeus, 1758, or Crocodylus niloticus Laurenti, 1768 than to Proterosuchus fergusi Broom, 1903a or Erythrosuchus africanus Broom, 1905 (stem-based).

Temporal range. Late Early Triassic (latest Olenekian, Ctenosauriscus koeneni; Butler et al., 2011) to Recent (Passer domesticus).

Diagnosis (synapomorphies). Parabasisphenoid without intertuberal plate (237:1→0); femur with well ossified proximal articular surface, being flat or convex (491:1→0); femoral head anteromedially oriented (493:0→1); and femur with a crest-like attachment of the caudofemoralis musculature on the posterior surface of the bone, without intertrochanteric fossa (=fourth trochanter), and not convergent with proximal end (504:0/1→2).

Remarks. The results of the phylogenetic analysis of this study bolster the non-monophyly of “Proterosuchia” and the presence of several successive sister-taxa of Euparkeria capensis and more crownward archosauriforms. Here at least eight lineages of non-archosaurian archosauriforms are recognized, including five branches of proterosuchian-grade species. Therefore, it is considered here useful to erect the new suprageneric taxon Eucrocopoda with the aim of including non-proterosuchian archosauriforms. Eucrocopoda means “true crocodile foot” (see above) and refers to the fact that members of this clade possess a series of character-states present in crocodile-line Archosauria but absent in non-eucrocopodan crocopods, such as a well ossified femoral head and a fourth trochanter.

Unnamed clade (Euparkeria capensis+ Archosauria)

Temporal range. Late Early Triassic (latest Olenekian, Ctenosauriscus koeneni; Butler et al., 2011) to Recent (Passer domesticus).

Synapomorphies. Basal tubera of the basioccipital clearly separated from each other (227:1→0); and external foramina for the passage of the abducens nerves (CN VI) open dorsally (251:1→0).

Synapomorphy present in only some trees. Femur with distal transverse width versus total length = 0.13–0.24 (510:2→1).

Unnamed clade (Dongusuchus efremovi+Yarasuchus deccanensis)

Temporal range. Early Middle Triassic (Anisian, Dongusuchus efremovi and Yarasuchus deccanensis; Lucas, 2010).

Synapomorphies. Femur with a transverse groove on the proximal surface (495:0→1); and femoral distal articular surface with both condyles do not projecting distally (distal articular surface concave or almost flat) (512:1→2).

Remarks. Niedźwiedzki, Sennikov & Brusatte (2014) described the presence of a low “anterior trochanter-like” structure on the anterior surface of the proximal end of the femur of Dongusuchus efremovi. A topologically and morphologically similar but better developed trochanter is present on several femora of Yarasuchus deccanensis (ISI R334) and may also represent an apomorphy of this clade.

Unnamed clade (Proterochampsia + Archosauria)

Temporal range. Late Early Triassic (latest Olenekian, Ctenosauriscus koeneni; Butler et al., 2011) to Recent (Passer domesticus).

Synapomorphies. Absence of excavation immediately lateral to the base of postaxial cervical neural spines (337:1→0); absence of dorsally opening pit lateral to the base of the dorsal neural spines (361:1→0); and large biceps process on the lateral surface of the coracoid (401:0→1).

Synapomorphies present in only some trees. Posterior process of the jugal does not form entirely or almost entirely the ventral border of the infratemporal fenestra (103:1→0); postparietal absent as a separate ossification (171:0/1→2); absence of pseudolagenar recess between the ventral surface of the ventral ramus of the opisthotic and the basal tubera (223:0→1); parabasisphenoid horizontal (235:1→0); minimum height of the dentary versus length of the alveolar margin (including edentulous anterior end if present) = 0.05–0.14 (266:1→0); and absence of postaxial cervical intercentra (346:0→1).

Proterochampsia Bonaparte, 1971

Definition. The most inclusive clade containing Proterochampsa barrionuevoi Reig, 1959, but not Euparkeria capensis Broom, 1913, Erythrosuchus africanus Broom, 1905, Passer domesticus Linnaeus, 1758, or Crocodylus niloticus Laurenti, 1768 (stem-based) (Nesbitt, 2011).

Temporal range. Late Middle Triassic (Ladinian, Jaxtasuchus salomoni; Schoch & Sues, 2014) to Late Triassic (late Norian, Doswellia sixmilensis; Heckert, Lucas & Spielmann, 2012).

Synapomorphies. External naris dorsally directed (11:0→1); dorsoventral height of the snout at the level of the anterior tip of the maxilla versus dorsoventral height at the level of the anterior border of the orbit = 0.38–0.53 (21:2→1); lateral margin of the snout anterior to the prefrontal formed by the nasal and maxilla with gently rounded transition along the maxilla from the lateral to dorsal side of rostrum (23:0→1); maxilla contacts prefrontal (61:0→1); lateral row of teeth on the ventral surface of the anterior ramus (=palatal ramus) of the pterygoid raised on a thick, posteromedially-to-anterolaterally oriented ridge (198:0→1); tooth bearing portion of the dentary ventrally curved or deflected (267:1→2); cervical and anterior dorsal vertebrae without posterior centrodiapophyseal lamina (316:1→0); posterior cervical and anterior dorsal vertebrae without prezygodiapophyseal lamina (317:1→0); posterior cervical and anterior dorsal vertebrae without postzygodiapophyseal lamina (318:1→0); transverse width of conjoined pubic aprons versus total length of the bone = 1.48–1.94 (478:2→3); tibia with straight cnemial crest (517:0→1); metatarsal V without a hook-shaped proximal end (577:1→0); and phalanges on pedal digit V present and with a “poorly” developed first phalanx (582:0→1).

Doswelliidae Weems, 1980

Definition. The most inclusive clade that contains all archosauromorphs more closely related to Doswellia kaltenbachi Weems, 1980 than to Proterochampsa barrionuevoi Reig, 1959, Erythrosuchus africanus Broom, 1905, Caiman latirostris Daudin, 1802, or Passer domesticus Linnaeus, 1758 (stem-based) (Desojo, Ezcurra & Schultz, 2011).

Temporal range. Late Middle Triassic (Ladinian, Jaxtasuchus salomoni; Schoch & Sues, 2014) to Late Triassic (late Norian, Doswellia sixmilensis; Heckert, Lucas & Spielmann, 2012).

Synapomorphies. Posterior extension of the maxilla level with or posterior to posterior orbital border in mature individuals (71:1→0); medial process of the squamosal long, forming entirely or almost entirely the posterior border of the supratemporal fenestra (140:0→1); posterior surface of the supraoccipital with a prominent median, vertical peg (210:0→1); total length of the femur versus total length of the humerus = 1.62–1.74 (489:1→2); proximal portion of the fibula symmetrical or nearly symmetrical in lateral view (527:1→0); metatarsal V with a prominent and pointed outer process on the proximal lateral margin (578:0→1); more than two rows of dorsal osteoderms (588:1→3); presacral dorsal osteoderms square-shaped (595:1→0); and presence of appendicular osteoderms (599:0→1).

Unnamed clade (Doswellia kaltenbachi+Tarjadia ruthae)

Temporal range. Late Middle Triassic (Ladinian, Jaxtasuchus salomoni; Schoch & Sues, 2014) to Late Triassic (late Norian, Doswellia sixmilensis; Heckert, Lucas & Spielmann, 2012).

Synapomorphies. Retroarticular process of the surangular and articular not upturned (284:1→0); transverse processes in middle and posterior dorsal vertebrae extremely long, being considerably broader than their respective centra (358:1→2); humerus with a medially expanded, asymmetric proximal end in anterior view (419:0→1); internal tuberosity of the humerus distinctly separated from the proximal articular surface of the bone (421:0→1); ilium with a laterally deflected iliac blade (459:0→1); ilium with convex dorsal margin of the iliac blade (466:1→0); external surface of the osteoderms sculptured (589:0→1); and unornamented anterior articular lamina in paramedian osteoderms (596:0→1).

Unnamed clade (Jaxtasuchus salomoni+Doswellia kaltenbachi)

Temporal range. Late Middle Triassic (Ladinian, Jaxtasuchus salomoni; Schoch & Sues, 2014) to Late Triassic (late Norian, Doswellia sixmilensis; Heckert, Lucas & Spielmann, 2012).

Synapomorphies. Basioccipital with notochordal scar on the occipital surface of the occipital condyle developed as a vertical furrow or a large sub-circular fossa that occupies approximately half of the height of the condyle surface (230:0→1); and blunt, anteroposteriorly restricted eminence on the external surface of paramedian osteoderms (591:0→2).

Unnamed clade (Tarjadia ruthae+Archeopelta arborensis)

Temporal range. Late Middle–early Late Triassic (Ladinian–early Carnian, Tarjadia ruthae and Archeopelta arborensis; Desojo, Ezcurra & Schultz, 2011).

Synapomorphies. Very thick paramedian osteoderms (592:0→1); and presacral dorsal osteoderms wider than long (595:0→2).

Proterochampsidae Sill, 1967

Definition. The least inclusive group that is composed of Chanaresuchus bonapartei Romer, 1971a and Proterochampsa barrionuevoi Reig, 1959 but not Euparkeria capensis Broom, 1913, Doswellia kaltenbachi Weems, 1980, Passer domesticus Linnaeus, 1758 nor Crocodylus niloticus Laurenti, 1768 (node-based) (Trotteyn, 2011b).

Temporal range. Early Late Triassic (early Carnian, Chanaresuchus bonapartei; Marsicano et al., 2015) to middle Late Triassic (earliest Norian, Proterochampsa barrionuevoi; Martínez et al., 2011).

Synapomorphies. Dorsal orbital margin with a shelf/ridge elevated above skull table that extends along the lateral surface of the lacrimal, prefrontal, frontal and postorbital (7:0→2); premaxilla with a slightly downturned main body, in which the alveolar margin is angled at approximately 20°to the alveolar margin of the maxilla (29:0→1); narial fossa expanded in the anteroventral corner of the naris (32:0→1); basioccipital with notochordal scar on the occipital surface of the occipital condyle developed as a vertical furrow or a large sub-circular fossa that occupies approximately half of the height of the condyle surface (230:0→1); parabasisphenoid with anteriorly arched intertuberal plate (237:0→2); ventral surface of the centrum in anterior cervical vertebrae with a median longitudinal keel that extends ventral to the centrum rim in at least one anterior cervical (327:1→2); and lateral posterior condyle of the proximal end of the tibia offset anteriorly from the medial posterior condyle (519:1→0).

Proterochampsa (Proterochampsa barrionuevoi+Proterochampsa nodosa)

Temporal range. Early to middle Late Triassic (late Carnian to middle Norian, Proterochampsa barrionuevoi; Martínez et al., 2011).

Synapomorphies. Strongly dorsoventrally compressed skull with mainly dorsally facing antorbital fenestrae and orbits (3:0→1); alveolar margin of the premaxilla does not reach the contact with the maxilla and forms a diastema (=subnarial gap) (26:0→1); posterior extension of the maxilla level with or anterior to anterior orbital border in mature individuals (71:1→2); tooth bearing portion of the dentary mostly straight (267:2→0); retroarticular process of the lower jaw absent (283:1→0); and posteroventral surface of the angular ridged or keeled (292:1→0).

Unnamed clade (Cerritosaurus binsfeldi+ Rhadinosuchinae)

Temporal range. Early Late Triassic (early Carnian, Chanaresuchus bonapartei; Marsicano et al., 2015) to middle Late Triassic (earliest Norian, Rhadinosuchus gracilis; Martínez et al., 2011).

Synapomorphies. Anterior margin of the antorbital fenestra nearly pointed (14:0→1); longitudinal ridge on the lateral surface of the main body of the jugal (97:0→1); depression on the lateral surface of the ventral process of the postorbital (133:0→1); anterior process of the squamosal forms most of the lateral border of the supratemporal fenestra (137:0→1); posterodorsal portion of the squamosal with a supratemporal fossa (149:0→1); widely concave notch on the anterior margin of the ascending process of the quadratojugal (154:0→1); and anteroposterior length of the external mandibular fenestra versus anteroposterior length of the dentary anterior to the fenestra = 0.44–0.53 (263:0→1).

Unnamed clade (Tropidosuchus romeri+ Rhadinosuchinae)

Temporal range. Early Late Triassic (early Carnian, Chanaresuchus bonapartei; Marsicano et al., 2015) to middle Late Triassic (earliest Norian, Rhadinosuchus gracilis; Martínez et al., 2011).

Synapomorphies. Supratemporal fossa immediately medial or anterior to the supratemporal fenestra on the dorsal surface of the skull roof (8:0→1); antorbital fossa absent from the lateral surface of the maxilla (54:1→0); infratemporal fossa marked by a sharp edge on the quadratojugal (152:0→1); suture between surangular and angular elevated and separates dorsal concave area on surangular from concave area on angular (281:0→1); and ventrolateral surface of the angular with a laterally projecting ridge that separates the lateral and ventral sides of the angular (291: 0→1).

Rhadinosuchinae Hoffstetter, 1955

Definition. All archosauriforms more closely related to Rhadinosuchus gracilis Huene, 1938 and Chanaresuchus bonapartei Romer, 1971a than to Cerritosaurus binsfeldi Price, 1946, Tropidosuchus romeri Arcucci, 1990, and Doswellia kaltenbachi Weems, 1980 (stem-based) (Ezcurra, Desojo & Rauhut, 2015).

Temporal range. Early Late Triassic (early Carnian, Chanaresuchus bonapartei; Marsicano et al., 2015) to middle Late Triassic (earliest Norian, Rhadinosuchus gracilis; Martínez et al., 2011).

Synapomorphies. Dorsal surface of the nasals and/or frontals ornamented by ridges radiating from centres of growth (6:0→1); lateral margin of the snout anterior to the prefrontal formed by the nasal and maxilla with sharp edge along the maxilla between the lateral and dorsal sides of this bone (=box-like snout of Kischlat, 2000) (23:1→2); anterior process of the jugal subrectangular or slightly dorsoventrally expanded, being higher than the portion of the maxilla underneath it (92:0→1); anteriorly projecting, rounded spur on the anterior edge of the ventral process of the postorbital indicating the lower delimitation of the eye-ball (134:0→1); and very thick paramedian osteoderms (592:0→1).

Unnamed clade (Gualosuchus reigi+Chanaresuchus bonapartei)

Temporal range. Early Late Triassic (early Carnian, Chanaresuchus bonapartei; Marsicano et al., 2015) to middle Late Triassic (earliest Norian, Rhadinosuchus gracilis; Martínez et al., 2011).

Synapomorphies. Posterior process of the squamosal ventrally curved (142:0→1); proximal end of the humerus medially expanded, being asymmetric in anterior view (419:0→1); and fourth trochanter of the femur distally extended beyond mid-shaft and well posteriorly developed (507:0→1).

Unnamed clade (Chanaresuchus bonapartei+Rhadinosuchus gracilis)

Temporal range. Early Late Triassic (early Carnian, Chanaresuchus bonapartei; Marsicano et al., 2015) to middle Late Triassic (earliest Norian, Rhadinosuchus gracilis; Martínez et al., 2011).

Synapomorphies. Antorbital fossa forming a distinct inset margin to the antorbital fenestra on the lateral surface of the lacrimal and occuping almost half or more of the anteroposterior length of the ventral process of the lacrimal (88:0→2); and squared presacral dorsal osteoderms (595:1→0).

Archosauria Cope, 1869–1870

Definition. The least inclusive clade containing Crocodylus niloticus Laurenti, 1768, and Passer domesticus Linnaeus, 1758 (node-based) (Sereno, 2005).

Temporal range. Late Early Triassic (latest Olenekian, Ctenosauriscus koeneni; Butler et al., 2011) to Recent (Passer domesticus).

Synapomorphies. Maxilla with an antorbital fossa on the ascending and horizontal processes of the maxilla (but not reaching the posteroventral corner of the fenestra in basal forms) (54:1→2); supratemporal fossa extending onto the ascending process of the postorbital (129:0→1); vomer without teeth (187:2→3); palatine and ventral surface of the anterior ramus of the pterygoid without teeth (188:0→1); exoccipital with a clear crest (=metotic strut) present on the lateral surface of the bone posterior to the external foramina for hypoglossal nerve (CN XII) (220:0→1); foramina for entrance of the cerebral branches of the internal carotid artery anterolaterally placed on the parabasisphenoid (240:0→2); anterior margin of the scapula and coracoid at level of the suture between both bones without a notch (385:1→0); humerus with a medially expanded proximal end, being asymmetric in anterior view (419:0→1); internal tuberosity of the humerus distinctly separated from the proximal articular surface (421:0→1); olecranon process of the ulna prominent but lower than its anteroposterior depth at base (430:0→1); femoral head with a posteromedial tuber (=anteromedial tuber of Nesbitt, 2011) (496:0→1); ventral articular surface of the calcaneum for distal tarsal 4 and the distal end of the calcaneal tuber separated by a clear gap (552:0→1); articular surfaces for fibula and distal tarsal 4 in the calcaneum continuous (555:0→1); and relation between paramedian dorsal osteoderms and presacral vertebrae one to one (593:1→0).

Pseudosuchia von Zittel, 1887–1890

Definition. The most inclusive clade containing Crocodylus niloticus Laurenti, 1768, but not Passer domesticus Linnaeus, 1758 (stem-based) (Sereno, 2005).

Temporal range. Late Early Triassic (latest Olenekian, Ctenosauriscus koeneni; Butler et al., 2011) to Recent (Crocodylus niloticus).

Synapomorphies. Posttemporal fenestra smaller than the supraoccipital but does not develop as a small foramen (19:2→1); posterior process of the squamosal ventrally curved (142:0→1); anterior ramus (=palatal process) of the pterygoid transversely narrow along its entire extension, converging in a right or acute angle with the transverse ramus, with the bone possessing an overall L-shape contour in ventral or dorsal view (194:0→1); posteroventral process of the dentary contributes to the border of the external mandibular fenestra (275:1→2); articular with a ventromedially directed process (295:0→1); spine tables on the distal ends of the postaxial cervical and dorsal neural spines (321:0→3); distal articular surface of the femur uneven, fibular condyle projecting distally distinctly beyond tibial condyle (512:1→0); area of attachment of the iliofibularis muscle in the fibula placed on a hypertrophied tubercle (529:0/1→2); calcaneal tuber approximately as broad as or broader than tall at midshaft (547:0→1); calcaneal tuber with a expanded distal end in proximal or distal view (549:0→1); transverse width of the distal articular surface of the calcaneum versus transverse width of the astragalus 0.54–1.22 (556:1→2); pedal unguals strongly transversely compressed, with a sharp dorsal keel (586:0→2); and presence of two rows of dorsal osteoderms (588:1→2).

Phytosauria Meyer, 1861

Definition. The most inclusive clade containing Rutiodon carolinensis Emmons, 1856 but not Aetosaurus ferratus Fraas, 1877, Rauisuchus tiradentes Huene, 1942, Prestosuchus chiniquensis Huene, 1942, Ornithosuchus longidens Huxley, 1877, and Crocodylus niloticus Laurenti, 1768 (stem-based) (Sereno, 2005).

Temporal range. Early Late Triassic (late Carnian–earliest Norian, Parasuchus hislopi; Langer, 2005) to latest Triassic (late Norian–?Rhaetian, Redondasaurus; Hunt & Lucas, 1993).

Synapomorphies. Strongly dorsoventrally compressed skull with mainly dorsally facing antorbital fenestrae and orbits (3:0→1); dorsal orbital margin of the frontal elevated above skull table (7:0→1); external naris non-terminal, considerably posteriorly displaced, but posterior rim of the naris well anterior to the anterior border of the orbit (10:0→1); external naris dorsally directed (11:0→1); orbit without or with incipient elevated rim (17:1→0); antorbital length versus total length of the skull = 0.70–0.76 (20:1→2); snout transversely broader than or as broad as dorsoventrally tall at the level of the anterior border of the orbit (22:1→0); 10 or more tooth positions in the premaxilla (42:1/2→0); maxilla extends anterior to the nasal (49:0→1); alveolar margin of the maxilla sigmoid, anteriorly concave and posteriorly convex, in lateral view (68:0→2); posterior extension of the maxilla level with or anterior to anterior orbital border in mature individuals (71:1→2); number of maxillary tooth positions 15–22 (75:1→2); lower temporal bar with a concave ventral margin in lateral view, though nowhere dorsal to tooth row (91:0→1); length of the posterior process of the jugal versus the height of its base = 4.07–5.37 (100:1→2); posterior process of the jugal lies ventral to the anterior process of the quadratojugal (105:0→1); ventral process of the postorbital ends close to or at the ventral border of the orbit (131:0→1); neomorphic bone as a separate ossification anterior to nasals and surrounded by the premaxilla on the dorsal surface of the snout (175:0→1); dorsal head of the quadrate has a sutural contact with the paroccipital process of the opisthotic (178:0→1); palatine with a single anterior process forming the posterior border of the choana (191:1→2); foramina for the entrance of the cerebral branches of the internal carotid artery leading to the pituitary fossa posterolaterally placed on the parabasisphenoid (240:2→1); mandibular symphysis present along one-third of the lower jaw (265:0→1); tooth-bearing portion of the dentary mostly straight (267:1→0); dorsal margin of the anterior portion of the dentary dorsally expanded compared to the dorsal margin of the posterior portion (271:0→1); posteriormost dentary teeth placed on the posterior half of lower jaw (277:0→1); distal edge of the posterior maxillary tooth crowns with a distinct different morphology from those of the anterior tooth crowns, with the posterior edge usually convex in labial view (302:0→1); scapula with an acromion process in about the same plane as the ventral edge of the scapula (395:1→0); coracoid with a distinctly hooked anterior border in lateral view (397:0→1); coracoid without or small biceps process on the lateral surface (401:1→0); humerus with a strongly developed entepicondyle in mature individuals (0); (425:0→1); pubic shaft rod-like and straight in lateral view (476:1→2); and osteoderms with sculpture on their external surface (589:0→1).

Parasuchus (Parasuchus hislopi+Parasuchus angustifrons)

Temporal range. Early Late Triassic (late Carnian–earliest Norian, Parasuchus hislopi; Langer, 2005).

Synapomorphies. Skull dermal sculpturing represented by prominent tubercles on the frontals, parietals, and nasals (5:1→2); jugal with bumps on the lateral surface of the main body (97:0→1); squamosal with an anterior process that forms most of the lateral border of the supratemporal fenestra (137:0→1); and posterodorsal portion of the squamosal with a supratemporal fossa (149:0→1).

Pseudopalatinae Long & Murry, 1995

Definition. The last common ancestor of Nicrosaurus kapffi Meyer, 1860, Mystriosuchus westphali Hungerbühler & Hunt, 2000, Machaeroprosopus pristinus (Mehl, 1928), Redondasaurus gregorii Hunt & Lucas, 1993, and all descendants of that ancestor (node-based) (Parker & Irmis, 2006).

Temporal range. Middle Late Triassic (Norian, Nicrosaurus kapffi; Stocker & Butler, 2013) to latest Triassic (late Norian–?Rhaetian, Redondasaurus; Hunt & Lucas, 1993).

Synapomorphies. Postnarial process of the premaxilla wide, plate-like (37:1→0); antorbital fossa on the lateral surface of the maxilla absent or not exposed in lateral view (54:2→0); posterior process of the postorbital extends close to or beyond the level of the posterior margin of the supratemporal fenestrae (130:0→1); squamosal with an anteroventrally directed ventral process at 45°or less (145:0→1); subtriangular quadratojugal (151:0→1); pterygoid with a posterolaterally oriented lateral ramus, forming an obtuse angle with the anterior ramus (200:1→0); exoccipital without a subvertical crest on its lateral surface (220:1→0); distinct dorsal process behind the alveolar margin of the lower jaw formed by a dorsally well-developed surangular (261:0→1); anteroposterior length of the external mandibular fenestra versus anteroposterior length of the dentary anterior to the fenestra = 0.44–0.53 (263:0→1); femoral head without a posteromedial tuber (=anteromedial tuber of Nesbitt, 2011) (496:1→0); and fibula with the attachment site of the iliofibularis muscle near the midpoint between the proximal and distal ends (530:0→1).

Unnamed clade (Nundasuchus songeaensis+ Suchia)

Temporal range. Late Early Triassic (latest Olenekian, Ctenosauriscus koeneni; Butler et al., 2011) to Recent (Crocodylus niloticus).

Synapomorphies. Coracoid with postglenoid process, separated from the glenoid fossa by a notch (402:0→1); proximal surface of the lateral condyle of the tibia depressed (518:0→1); proximal surface of the distal tarsal 4 with a distinct, proximally raised region on the posterior portion (=heel of Sereno & Arcucci, 1994a) (562:0→1); and paramedian osteoderms with a longitudinal keel on its external surface, extending along all or most of the anteroposterior length of the osteoderm as a transversely compressed flange (591:0→1).

Unnamed clade (Ornithosuchidae + Suchia)

Temporal range. Late Early Triassic (latest Olenekian, Ctenosauriscus koeneni; Butler et al., 2011) to Recent (Crocodylus niloticus).

Synapomorphies. Transverse width of conjoined pubic aprons versus total length of the bone = 0.27–0.97 (478:2→0/1); minimum transverse width of the femur versus minimum transverse width of the humerus = 1.46–1.80 (490:1→2); fibula with attachment site of the iliofibularis muscle near the midpoint between the proximal and distal ends (530:0→1); tibial facet of the astragalus divided into distinct posteromedial and anterolateral basins (536:0→1); and calcaneal tuber posteriorly oriented between 71°–90°(546:1→2).

Ornithosuchidae Huene, 1908

Definition. Ornithosuchus longidens (Huxley, 1877), Riojasuchus tenuisceps Bonaparte, 1967, Venaticosuchus rusconii Bonaparte, 1970 and all descendants of their most recent common ancestor (node-based) (Sereno, 1991).

Temporal range. Early Late Triassic (late Carnia–earliest Norian, Venaticosuchus rusconi; Martínez et al., 2011) to middle Late Triassic (middle Norian, Riojasuchus tenuisceps; Kent et al., 2014).

Synapomorphies. Alveolar margin of the premaxilla does not reach the contact with the maxilla and forms a diastema (=subnarial gap) (26:0→1); slightly downturned main body of the premaxilla, alveolar margin is angled at approximately 20°to the alveolar margin of the maxilla (29:0→1); postnarial process of the premaxilla overlaps the anterodorsal surface of the nasal (39:0→1); three tooth positions in the premaxilla (42:1/2→3); anteroposterior length of the antorbital fossa anterior to the antorbital fenestra versus length of the antorbital fenestra = 0.28–0.43 (55:0→1); eight or nine tooth positions in the maxilla (75:1→0); anterior process of the quadratojugal distinctly present, in which the lower temporal bar is complete and participates in the posteroventral border of the infratemporal fenestra, and process finishes close to the base of the posterior process of the jugal (153:2→3); parietal extends over interorbital region (160:0→1); lateral ramus of the pterygoid posterolaterally oriented, forming an obtuse angle with the anterior ramus (200:1→0); mandibular symphysis present along one-third of the lower jaw (265:0→1); tooth-bearing portion of the dentary ventrally curved or deflected (267:1→2); surangular with a laterally projecting shelf that possesses a strongly convex lateral edge (286:2→3); three sacral vertebrae (370:0→1); coracoid with a subglenoid lip more posteriorly extended than the supraglenoid lip on the scapula (399:0→1); ectepicondylar region of the humerus without supinator process, groove or foramen (427:1→2); perforated acetabulum (455:0→1); total length of the pubis versus anteroposterior length of the acetabulum = 2.84–3.43 (472:0→1); anterior and posterior portions of the acetabular margin of the pubis recessed (473:0→1); ischium with articular surfaces with the ilium and pubis separated by a non-articulating concave surface (483:0→2); femur with anterior trochanter (502:0→1); articulation between astragalus and calcaneum concavoconvex with concavity on the astragalus (532:1→2); and fibular facet of the calcaneum slightly convex (553:1→0).

Suchia Krebs, 1974

Definition. The least inclusive clade containing Aetosaurus ferratus Fraas, 1877, and Rauisuchus tiradentes Huene, 1942, Prestosuchus chiniquensis Huene, 1942, Crocodylus niloticus Laurenti, 1768 (node-based) (Nesbitt, 2011).

Temporal range. Late Early Triassic (latest Olenekian, Ctenosauriscus koeneni; Butler et al., 2011) to Recent (Crocodylus niloticus).

Synapomorphies. Dorsal vertebrae with well-rimed lateral fossa on the centrum below the neurocentral suture (354:1→2); proximal end of the humerus approximately symmetric in anterior view (419:1→0); and preacetabular process of the ilium longer than two thirds of its height, but not extending beyond the level of the anterior margin of the pubic peduncle (460:1→2).

Synapomorphies present in only some trees. Triangular dorsal process with clear dorsal apex formed by discrete expansion of the posterior end of the horizontal process of the maxilla in lateral view (65:0→1); and calcaneal tuber midshaft just less than twice the transverse width of the fibular facet (547:1→2).

Unnamed clade (Gracilisuchidae + Loricata)

Temporal range. Late Early Triassic (latest Olenekian, Ctenosauriscus koeneni; Butler et al., 2011) to Recent (Crocodylus niloticus).

Synapomorphy. Presacral paramedian osteoderms with a distinct longitudinal bend near the lateral edge (598:0→1).

Synapomorphies present in only some trees. Dorsal surface of the temporal region of the skull with a supratemporal fossa immediately medial or anterior to the supratemporal fenestra (8:0→1); anteroposterior length of the main body of the premaxilla versus its maximum dorsoventral height = 1.07–2.00 (28:2→1); alveolar margin on the anterior third of the maxilla (anterior to the level of the anterior border of the antorbital fenestra if present) distinctly upturned (70:0→1); dentary with posterocentral process, in which its margins are not confluent with the dorsal or ventral margin of the lower jaw (273:0→1); chevrons with distal anteroposterior width of the anterior and middle haemal spines equivalent to proximal width in lateral view (382:1→0); scapula with lateral tuber on the posterior edge, just dorsal of the glenoid fossa (393:0→1); and calcaneum with ventral articular surface for distal tarsal 4 and the distal end of the calcaneal tuber separated by a clear gap (552:1→2).

Gracilisuchidae Butler et al., 2014a

Definition. The most inclusive clade containing Gracilisuchus stipanicicorum Romer, 1972c but not Ornithosuchus longidens (Huxley, 1877), Aetosaurus ferratus Fraas, 1877, Poposaurus gracilis Mehl, 1915, Postosuchus kirkpatricki Chatterjee, 1985, Rutiodon carolinensis Emmons, 1856, Erpetosuchus granti Newton, 1894, Revueltosaurus callenderi Hunt, 1989, Crocodylus niloticus (Laurenti, 1768), or Passer domesticus Linnaeus, 1758 (stem-based) (Butler et al., 2014a).

Temporal range. Early Middle Triassic (Anisian, Turfanosuchus dabanensis) to early Late Triassic (earliest Carnian, Gracilisuchus stipanicicorum; Marsicano et al., 2015) (Butler et al., 2014a).

Synapomorphies. Postnarial process of the premaxilla fits into slot of the nasal (39:0→2); length of the portion of the maxilla anterior to the antorbital fenestra versus the total length of the bone = 0.12–0.22 (50:1→0); participation of the nasal in the dorsal border of the antorbital fossa (83:0→1); anterior process of the lacrimal forms the entire or almost the entire dorsal border of the antorbital fenestra (87:0→1); ventral process of the squamosal anteroventrally directed at 45°or less (145:0→1); anterior and middle postaxial cervical neural spines with an anterior overhang (343:0→1); cervical and dorsal vertebrae with fan-shaped neural spine in lateral view (363:0→1); and internal tuberosity of the humerus not distinctly separated from the proximal articular surface (421:1→0).

Prestosuchus chiniquensis+Batrachotomus kupferzellensis (?=Paracrocodylomorpha)

Temporal range. Late Early Triassic (latest Olenekian, Ctenosauriscus koeneni; Butler et al., 2011) to Recent (Crocodylus niloticus).

Synapomorphies. Posttemporal fenestra developed as a small foramen (19:1→2); maxilla with palatal process distinctly dorsal to the bases of the interdental plates (67:0→1); total length of the nasal versus total length of the frontal = 2.26–3.09 (76:1→2); exposure of the lacrimal on the skull roof absent or marginal in dorsal view (86:1→0); frontal orbital border absent or anteroposteriorly short in mature individuals (114:1→0); minimum height of the dentary versus length of the alveolar margin (including edentulous anterior end if present) = 0.16–0.29 (266:0→1/2); tooth bearing portion of the dentary mostly straight (267:1→0); dentary with dorsal margin of the anterior portion dorsally expanded, when compared to the dorsal margin of the posterior portion (271:0→1); middle and posterior dorsal vertebrae with hyposphene-hypantrum accessory intervertebral articulation (359:0→1); ulna with subrectangular or slightly expanded olecranon process towards the proximal tip of the bone (431:0→1); ilium with posteriorly projected heel on the posterior margin of the ischiadic peduncle in lateral view (468:0→1); pubic-ischium suture reduced to a thin proximal contact (470:0→1); pubis with a sharply anteroposteriorly expanded distal end in lateral or medial view, forming a distinct pubic boot (480:0→1); ischium with an extensive contact with antimere, but the dorsal margins are separated (485:0→1); femur with transverse groove on the proximal surface (495:0→1); and metatarsal V with dorsal prominence separated from the proximal surface by a concave gap (576:0→1).

Unnamed clade (Batrachotomus kupferzellensis+Youngosuchus sinensis)

Temporal range. Middle Triassic (Anisian–Ladinian, Youngosuchus sinensis; Lucas, 2010).

Synapomorphies. Premaxilla with a short postnarial process that ends well anterior to the posterior margin of the external naris (36:2→1); maxilla with lateroventrally facing neurovascular foramina that extend ventrally as deep, well-defined grooves on the lateral surface of the anterior and horizontal processes (53:0→1); maxilla with a concave anterodorsal margin of the base of the ascending process (59:0→1); frontal with longitudinal ridge along the midline of the dorsal surface (115:0→1); and supraoccipital with a prominent median, vertical peg on the posterior surface (210:0→1).

Ornithodira Gauthier, 1986

Definition. The least inclusive clade containing Pterodactylus antiquus Sömmerring, 1812, and Passer domesticus Linnaeus, 1758 (node-based) (Nesbitt, 2011).

Temporal range. Early Middle Triassic (Anisian, Asilisaurus kongwe; Nesbitt et al., 2010) to Recent (Passer domesticus). The presence of probable dinosauromorph footprints in the Olenekian may pull back the temporal range of ornithodirans into the late Early Triassic (Brusatte, Niedźwiedzki & Butler, 2011).

Synapomorphies. Scapula and coracoid fused with each other without persistent suture in mature individuals (384:1→0); total length of the scapula versus minimum anteroposterior width of the scapular blade = 7.92–11.31 (387:0→1); coracoid with a subglenoid lip more posteriorly extended than the supraglenoid lip on the scapula (399:0→1); humerus without supinator process, groove or foramen in the ectepicondylar region (427:1→2); manual unguals of digits I–III trenchant (452:0→1); second phalanx of manual digit II longer than the first phalanx of manual digit II (453:0→1); pelvic girdle with acetabular antitrochanter (457:0→1); preacetabular process of the ilium longer than two thirds of its height and does not extend beyond the level of the anterior margin of the pubic peduncle (460:1→2); dorsal margin of the iliac blade concave (466:1→2); thin femoral bone wall thickness at or near midshaft, thickness/diameter <0.3 (508:0→1); anterior hollow of the astragalus reduced to a foramen (=extensor canal) or absent (538:0→1); astragalus without posterior groove (539:0→1); anteromedial corner of the astragalus acute in proximal view (540:0→1); calcaneum terminating in an unthickened lateral margin (544:1→0); calcaneal tuber absent or incipient (545:1→0); distal tarsal 4 transverse width subequal to distal tarsal 3 (560:0→1); distal tarsal 4 with articular facet for metatarsal V less than half of the lateral surface of the bone (561:0→1); compact metatarsus, metatarsals I–IV tightly bunched (565:0→1); and dorsal osteoderms absent (588:1→0).

Dinosauromorpha Benton, 1985

Definition. The most inclusive clade containing Passer domesticus Linnaeus, 1758, but not Pterodactylus antiquus Sömmerring, 1812, Ornithosuchus longidens Huxley, 1877, Crocodylus niloticus Laurenti, 1768 (stem-based) (Sereno, 2005).

Temporal range. Early Middle Triassic (Anisian, Asilisaurus kongwe; Nesbitt et al., 2010) to Recent (Passer domesticus). The presence of probable dinosauromorph footprints in the Olenekian may pull back the temporal range of dinosauromorphs into the late Early Triassic (Brusatte, Niedźwiedzki & Butler, 2011).

Synapomorphies. Pubis with prominent tuberosity for the attachment of the ambiens muscle in mature individuals (474:1→0); proximal articular surface of the femur (=posterolateral portion of the head sensu Nesbitt, 2011) extends under the proximal surface of the bone (494:0→1); tibia with straight, distinct cnemial crest (517:0→1); metatarsal V without a hook-shaped proximal end (577:1→0); and pedal digit V absent (582:0→2).

Dinosauriformes Novas, 1992b

Definition. The least inclusive clade containing Passer domesticus Linnaeus, 1758, and Marasuchus lilloensis (Romer, 1972b) (node-based) (Sereno, 2005).

Temporal range. Early Middle Triassic (Anisian, Asilisaurus kongwe; Nesbitt et al., 2010) to Recent (Passer domesticus).

Synapomorphies. Suture between pubis and ischium reduced to a thin proximal contact (470:0→1); total length of the pubis versus anteroposterior length of the acetabulum = 2.84–3.43 (472:0→1); anterior and posterior portions of the acetabular margin of the pubis recessed (473:0→1); transverse width of conjoined pubic aprons versus total length of the bone = 0.27–0.59 (478:2→0); ischium with proximal articular surfaces with the ilium and pubis separated by a fossa (483:0→1); femur with trochanteric fossa on the posterior surface of the proximal end (499:1→0); femur with anterior trochanter (502:0→1); femur with trochanteric shelf (503:0→1); femur without anterior extensor groove in the distal end (513:1→0); tibia with posterolateral process (=lateral malleolus) in the distal end (521:0→1); lateral side of the distal portion of the tibia with a proximodistally oriented groove (524:0→1); and astragalus with ascending process (537:0→1).

Unnamed clade (Silesauridae + Dinosauria)

Temporal range. Early Middle Triassic (Anisian, Asilisaurus kongwe; Nesbitt et al., 2010) to Recent (Passer domesticus).

Synapomorphies. Posterior surface of the supraocciptial with a prominent median, vertical peg (210:0→1); anterior tympanic recess on the lateral side of the braincase (245:0→1); atlantal articulation facet in the axial intercentrum concave with upturned lateral borders (325:0→1); total length of the pubis versus anteroposterior length of the acetabulum = 3.94–4.87 (472:1→2); ischium with longitudinal groove on the dorsal surface of shaft (484:0→1); extensive medial contact between ischia but the dorsal margins are separated (484:0→1); dorsolateral trochanter on the anterolateral surface of the proximal end of the femur (500:0→1); anterior edge of the proximal portion of the fibula tapers to a point and arched anteromedially (526:0→1); ascending process of the astragalus restricted to the anterior half of the astragalar depth (537:1→2); metatarsals I and V mid-shaft diameters lower than those of metatarsals II–IV (568:0→1); and distal articular surface of metatarsal IV as broad as deep as or deeper than broad (asymmetrical) (575:0→1).

Silesauridae Langer et al., 2010b

Definition. All archosaurs closer to Silesaurus opolensis Dzik, 2003, than to Heterodontosaurus tucki Crompton & Charig, 1962 and Marasuchus lilloensis (Romer, 1972b) (stem-based) (Langer et al., 2010b).

Temporal range. Early Middle Triassic (Anisian, Asilisaurus kongwe; Nesbitt et al., 2010) to late Late Triassic (middle Norian, Eucoelophysis baldwini; Irmis et al., 2011).

Synapomorphies. Ratio between transverse width of the diapophysis and length of the centrum in anterior dorsal vertebrae <0.70 (357:1→0).

Dinosauria Owen, 1842

Definition. The least inclusive clade containing Triceratops horridus (Marsh, 1889) and Passer domesticus Linnaeus, 1758 (node-based) (Sereno, 2005).

Temporal range. Early Late Triassic (late Carnian, Sanjuansaurus gordilloi; Martínez et al., 2011) to Recent (Passer domesticus).

Synapomorphies. Narial fossa of the premaxilla expanded in the anteroventral corner of the naris (32:0→1); postnarial process of the premaxilla wide, plate-like (37:1→0); frontal participates on the anteromedial corner of the supratemporal fossa (117:0→1); epipophyses on anterior postaxial cervical vertebrae (336:0→1); forelimb-hindlimb length ratio <0.55 (414:0→1); perforated acetabulum (455:0→1); main axis of the postacetabular process of the ilium mainly posteriorly oriented in lateral or medial views (464:0→1); pubic shaft vertically or posteroventrally oriented (475:0→1); ischial articular surface with the ilium and pubis separated by a non-articulating notched surface (483:1→2); fourth trochanter asymmetrical, with the distal margin forming a steeper angle to the shaft in medial or lateral view (506:0→1); cnemial crest of the proximal end of the tibia anterolaterally curved (517:1→2); posterior surface of the distal end of the tibia with a distinct proximodistally oriented ridge (522:0→1); and fibula with proximal portion symmetrical or nearly symmetrical in lateral view (527:1→0).

Homoplasy metrics

The population of all consistency indexes (CIs) show a non-normal distribution and the same occurs in eleven of the twelve anatomical groups (Shapiro–Wilk test, p < 0.001 for all the tests with the exception of the hindlimb, p = 0.010), in which only the values obtained for the pelvic girdle show a significant normal distribution (Shapiro–Wilk test, p = 0.097). Most of the values range 0.1–0.5 and the highest frequency is recovered in values between 0.45 and 0.50 (Fig. 57). The second highest peak is present in values between 0.95 and 1.00 and it is clearly separated from the distribution of the other values. As a result, there is a distinct dichotomy between highly homoplastic and poorly homoplastic characters in this data matrix and the CIs present a multimodal distribution (Hartigans’ dip test, p < 0.001). Indeed, the violin plot shows that most of the anatomical groups possess a bimodal or multimodal distribution (Fig. 58) and the Hartigans’ dip test found a significant multimodal distribution for all the anatomical groups (p < 0.05), with the exception of the forelimb (p = 0.09) and pelvic girdle (p = 0.25). Nine of the ten groups with a multimodal distribution possess two distinct peaks, one below or around 0.5 and the other between 0.85 and 1.00. The Kruskal–Wallis multiple comparison test failed to find significantly different means among the 12 groups, with the exception of a significant difference between the CI values of the lower jaw and hindlimb.

Figure 57 Bremer supports recovered after the pruning a posteriori of terminals with high amount of missing data in analysis 3.

Photograph of the lepidosauromorph and the extant dinosaur courtesy of Mariana Grassetti and photograph of the allokotosaurian courtesy of Sterling Nesbitt.

Figure 58 Histogram showing the distribution of consistency indexes recovered in analysis 3.

Discussion

Much of the general topology of the phylogenetic trees recovered in this analysis agrees with that found by several previous workers (e.g., Sereno, 1991; Dilkes, 1998; Gottmann-Quesada & Sander, 2009; Ezcurra, Lecuona & Martinelli, 2010; Nesbitt, 2011; Ezcurra, Scheyer & Butler, 2014). For example, tanystropheids, rhynchosaurs, and prolacertids are found as successive closer relatives of Archosauriformes, resembling the results of Dilkes (1998), Gottmann-Quesada & Sander (2009) and Ezcurra, Scheyer & Butler (2014), and proterosuchids and erythrosuchids are recovered as the least crownward archosauriforms (Figs. 48 and 49), in agreement with the topologies recovered by Ezcurra, Lecuona & Martinelli (2010) and Nesbitt (2011). However, some results recovered here differ from those recently found by other researchers (e.g., Euparkeria capensis as the sister-taxon of proterochampsids and archosaurs; phytosaurs within Pseudosuchia; Nesbitt, 2011). The main and more interesting results of this analysis are discussed as follows.

The phylogenetic position of Choristodera

The choristoderans are a group of freshwater aquatic diapsids known from the Jurassic to Miocene of the Northern Hemisphere (Matsumoto & Evans, 2010). The phylogenetic position of this group within Neodiapsida is highly controversial, being alternatively recovered outside Sauria (e.g., Dilkes, 1998; Gottmann-Quesada & Sander, 2009), as basal lepidosauromorphs (e.g., Müller, 2004; Bickelmann, Müller & Reisz, 2009) or as archosauromorphs (e.g., Gauthier, Kluge & Rowe, 1988; Neenan, Klein & Scheyer, 2013). Beyond the uncertain higher-level relationships of choristoderans, most analyses agree that the divergence of the group should have occurred during the middle Permian to earliest Triassic (e.g., Dilkes, 1998). As a result, there is a long ghost lineage between this inferred minimum divergence and the oldest unambiguous choristoderan (Cteniogenys sp. from the Middle Jurassic of Europe; see comments about Pachystropheus rhaeticus in the list of synapomorphies of Choristodera in Results). The problematic phylogenetic position of choristoderans may be a result of an unsampled early evolutionary history. The phylogenetic position of choristoderans is also ambiguously resolved in this analysis, but is constrained to the base of either Lepidosauromorpha or Archosauromorpha. Six synapomorphies support the position of choristoderans within Archosauromorpha and Lepidosauromorpha in each case (see Synapomorphies of some trees that include choristoderans in Results). It should be noted that only three additional steps are necessary to force the position of choristoderans outside Sauria, and the present taxonomic and character sample is not focused on non-archosauromorph diapsids. Therefore, this result invites future work testing the phylogenetic position of the group within Neodiapsida in light of the information provided here.

The phylogenetic position of Paliguana

Paliguana whitei has been historically considered an early member of the lizard lineage (e.g., Carroll, 1975) and more recently this hypothesis has been bolstered by quantitative phylogenetic analyses that found the species as the oldest and most basal lepidosauromorph (Evans & Borsuk-Białynicka, 2009; Ezcurra, Scheyer & Butler, 2014; Sobral, Sues & Müller, 2015). The results of the current analysis also support the position of Paliguana whitei as the earliest branching lepidosauromorph, which is supported by three craniodental characters. However, this result is not strongly supported and only two additional steps are necessary to force the position of Paliguana outside Lepidosauromorpha and place it as the sister-taxon of Sauria. The relatively weak support for the phylogenetic position of Paliguana whitei is not surprising because of the high amount of missing data that the terminal has in this analysis (approximately 85%; Table 2). Therefore, the finding of new, more complete specimens is necessary to shed light on the phylogenetic affinities of this species.

The phylogenetic position of Aenigmastropheus

The Permian archosauromorph record is very scarce and the only species of the group known from the Southern Hemisphere is the recently named Aenigmastropheus parringtoni from the late Permian of Tanzania (Ezcurra, Scheyer & Butler, 2014). Ezcurra, Scheyer & Butler (2014) recovered Aenigmastropheus parringtoni as the sister-taxon of Protorosaurus speneri, and as part of a “Protorosauria” also composed of the tanystropheids Macrocnemus bassanii and Tanystropheus longobardicus. The protorosaurs were found as the most basal archosauromorphs (Ezcurra, Scheyer & Butler, 2014) and these results have been recently also recovered by Sobral, Sues & Müller (2015) based on a slightly modified version of this original data matrix, in which the enigmatic Late Triassic diapsid Elachistosuchus huenei was added to the taxonomic sample. The phylogenetic analysis of Ezcurra, Scheyer & Butler (2014) was based on a modified version of that published by Reisz, Modesto & Scott (2011), which aimed to test if the supposed early diapsid Apsisaurus witteri (Laurin, 1991) was a pelycosaur or an early diapsid. As a result, Ezcurra, Scheyer & Butler (2014) pointed out that, although the character and taxonomic sampling of early archosauromorphs was improved with respect to that of Reisz, Modesto & Scott (2011), the topology that they recovered may be subject to considerable changes if more characters and non-archosauriform archosauromorphs were added. Furthermore, Ezcurra, Scheyer & Butler (2014) noted that the addition of more terminals may change the optimization of character-states recovered as synapomorphic of “Protorosauria” (e.g., laminae in the cervical and dorsal neural arches) and that Aenigmastropheus parringtoni would probably be recovered as the most basal archosauromorph.

Aenigmastropheus parringtoni was recovered as the most basal archosauromorph (with the exception of choristoderans in some trees) in this analysis, contrasting with the quantitative result of Ezcurra, Scheyer & Butler (2014), but in agreement with their prediction for the position of the species if included in an analysis sampling more early archosauromorphs. For example, the position of Aenigmastropheus parringtoni as the most basal archosauromorph implies that the presence of notochordal vertebrae is a condition retained from basal amniotes and not a feature convergently acquired with them or independently lost in other protorosaurs and archosauromorphs (contra Ezcurra, Scheyer & Butler, 2014). The position of this species within Archosauromorpha is supported by four vertebral characters, and two apomorphies exclude it from the clade formed by Protorosaurus speneri and more crownward archosauromorphs (see Results). Nevertheless, under constrained topologies, only one additional step is necessary to find Aenigmastropheus parringtoni as the sister-taxon of Protorosaurus speneri and within a monophyletic “Protorosauria” formed together with tanystropheids. Therefore, the phylogenetic relationships of non-crocopodan archosauromorphs should be considered weakly supported in this analysis and deserve further exploration and testing in future studies.

The non-monophyly of “Prolacertiformes”

Multiple gracile and long-necked Permian and Triassic archosauromorphs have been historically assigned to “Prolacertiformes,” and some quantitative phylogenetic analyses have supported the monophyly of the group (Benton & Allen, 1997; Jalil, 1997; Rieppel, Fraser & Nosotti, 2003). The most extensively taxonomically sampled quantitative analysis that tested the monophyly of “Prolacertiformes” was conducted by Jalil (1997). This analysis found 14 genera within “Prolacertiformes,” namely Protorosaurus, Prolacerta, Prolacertoides, Malutinisuchus, Kadimakara, Boreopricea, Malerisaurus, Jesairosaurus and six very likely aquatic genera that are usually grouped within Tanystropheidae (Trachelosaurus, Macrocnemus, Langobardisaurus, Cosesaurus, Tanystropheus, Tanytrachelos). However, more recent quantitative analyses failed to recover a monophyletic “Prolacertiformes” sensu Jalil (1997) and instead recovered Prolacerta broomi as more closely related to archosauriforms than to Protorosaurus speneri (Dilkes, 1998; Modesto & Sues, 2004; Gottmann-Quesada & Sander, 2009; Ezcurra, Scheyer & Butler, 2014).

The results of the current phylogenetic analysis support the closer position of Prolacerta broomi to Archosauriformes than to Protorosaurus speneri. Moreover, most of the putative prolacertiforms sampled by Jalil (1997) are recovered in a polyphyletic arrangement among non-archosauriform archosauromorphs (Fig. 53: red boxes). The only putative prolacertiforms (sensu Jalil, 1997) recovered here in a monophyletic group are Tanystropheus longobardicus and Macrocnemus bassanii, which are included within Tanystropheidae together with Amotosaurus rotfeldensis. Tanystropheids and Jesairosaurus lehmani are recovered more closely related to each other than to other archosauromorphs and as sister-taxa of Crocopoda. Prolacertoides jimusarensis is found as a member of the Jesairosaurus-tanystropheid clade or as an early allokotosaurian. Boreopricea funerea is recovered unambiguously within Crocopoda, as the sister-taxon of prolacertids and more crownward archosauromorphs.

Jalil (1997) recovered five synapomorphies for “Prolacertiformes” that are discussed below in the context of the present phylogenetic analysis.

(1) Skull low and narrow with short and narrow post-orbital region. This character-state is partially represented by characters 20 (antorbital length versus total length of the skull) and 22 (skull proportions at the level of the anterior border of the orbit) of the present phylogenetic analysis (Fig. 17). Putative prolacertiforms have an antorbital length that is 0.40–0.56 times the total length of the skull (e.g., Jesairosaurus lehmani: 0.47, ZAR 06; Macrocnemus bassanii: 0.56, PIMUZ T4822; Prolacerta broomi: 0.45, BP/1/471, SAM-PK-K10797) and the ratio present in immediate sister-taxa of Archosauromorpha (e.g., Youngina capensis: 0.49, GHG K 106) and non-eucrocopodan archosauriforms (Proterosuchus fergusi: 0.50–0.51, RC 846, SAM-PK-K11208) falls within this range. In addition, several basal archosauriforms also possess a narrow skull, being dorsoventrally taller than transversely broad at the level of the anterior border of the orbit (e.g., Proterosuchus fergusi: RC 846; Garjainia prima: PIN 2394/5-1). Thus, a low and narrow skull is plesiomorphic for archosauromorphs and not restricted to putative prolacertiforms.

(2) Low and elongate cervical neural spines. Anteroposteriorly long and dorsoventrally low cervical neural spines (character-state 342-1) are present in putative prolacertiforms (e.g., Protorosaurus speneri, Amotosaurus rotfeldensis, Macrocnemus bassanii, Tanystropheus longobardicus, Prolacerta broomi, Boreopricea funerea), whereas most non-archosaurian archosauriforms possess dorsoventrally taller cervical neural spines (e.g., Proterosuchus fergusi: BSPG 1934 VII 514; Proterosuchus alexanderi: NMQR 1484; Garjainia prima: PIN 2394/5; Euparkeria capensis: SAM-PK-5867). The present phylogenetic analysis optimizes the presence of low and elongate cervical neural spines as a symplesiomorphy of Archosauromorpha, which is retained by non-crocopodan archosauromorphs and subsequently lost in Trilophosaurus buettneri, rhynchosaurs and archosauriforms.

(3) Long and slender cervical ribs. The putative prolacertiforms Protorosaurus speneri, Amotosaurus rotfeldensis, Macrocnemus bassanii, and Tanystropheus longobardicus possess long cervical ribs, which are twice as long as their respective centra and oriented parallel to the neck (character-state 349-2). Nevertheless, non-archosaurian archosauriforms also possess this character-state (e.g., Proterosuchus alexanderi: NMQR 1484; Euparkeria capensis: SAM-PK-13665) and, as a result, it is optimized as an apomorphy of Archosauromorpha as a whole.

(4) Lacrimal does not meet the nasal. This character-state is linked with the contact between the maxilla and prefrontal (character-state 61-1) and is widely distributed among saurians, including early lepidosauromorphs (Evans, 1980; Fraser, 1982), choristoderans (Evans, 1990), rhynchosaurs (e.g., Mesosuchus browni: SAM-PK-6536; Rhynchosaurus articeps: NHMUK PV R1236), and several putative prolacertiforms (e.g., Tanystropheus longobardicus: Wild, 1973; Jesairosaurus lehmani: ZAR 08) (Fig. 17). By contrast, the lacrimal contacts the nasal in Prolacerta broomi (e.g., BP/1/471) and non-archosaurian archosauriforms (e.g., Proterosuchus fergusi: BSPG 1934 VII 514, RC 846; Erythrosuchus africanus: BP/1/5207; Euparkeria capensis: SAM-PK-5867). Therefore, the absence of a contact between the lacrimal and nasal is optimized as an apomorphy of Sauria, which is subsequently reversed in the clade composed of prolacertids and more crownward archosauromorphs.

(5) Loss of trunk intercentra. Most putative prolacertiforms lack intercentra in the trunk (e.g., Protorosaurus speneri: BSPG 1995 I 5, cast of WMsN P47361; Macrocnemus bassanii: PIMUZ T4822; Jesairosaurus lehmani: ZAR 11, 13), but they may occur in the posterior dorsal series of some tanystropheids (e.g., Amotosaurus rotfeldensis: SMNS 90600) (character-state 366-1). The optimization of the absence of trunk intercentra is ambiguous at the base of Archosauromorpha (beacause of the unknown condition in choristoderans), but their presence is reconstructed as the ancestral condition of Crocopoda. Therefore, the absence of dorsal intercentra in Protorosaurus speneri and tanystropheids is either a feature independently acquiered in both clades or retained from the ancestral condition of Archosauromorpha, and subsequently reacquired in crocopods.

The presence of low and elongate cervical neural spines and the absence of trunk intercentra are character-states that are consistent with the hypothesis of monophyly of Prolacertiformes (sensu Jalil, 1997), but the other three proposed synapomorphies are not consistent with this hypothesis. By contrast, multiple other character-states are shared by some putative prolacertiforms and more crownward archosauromorphs, but are absent in other putative prolacertiforms (e.g., premaxilla with a postnarial process that forms most of the border of the external naris, five or more tooth positions in the premaxilla, maxilla without contact with prefrontal, labiolingual compression of the marginal tooth crowns, pseudolagenar recess between the ventral surface of the ventral ramus of the opisthotic and the basal tubera, anterior and middle postaxial cervical neural spines without an anterior overhang, postaxial cervical vertebrae with epipophysis, interclavicle without anterior process). All these characters provide substantial evidence for the non-monophyly of “Prolacertiformes” sensu Jalil (1997) and 19 additional steps are necessary to recover the monophyly of the group under a topologically constrained search. In addition, some of the clades that include some putative prolacertiforms to the exclusion of others possess a moderately high Bremer support (=3; e.g., Crocopoda, Rhynchosauria + Archosauriformes, Boreopricea + Archosauriformes) following the pruning of taxa with a large amount of missing data (Fig. 59). As a result, the hypothesis of monophyly of “Prolacertiformes” sensu Jalil (1997) is rejected here. Instead, taxa previously identified as prolacertiforms form a polyphyletic assemblage distributed among the non-archosauriform archosauromorph region of the diapsid tree.

Figure 59 Violin box-plot showing the distribution, mean and standart deviation of the consistency indexes of each anatomical group in analysis 3.

The monophyly of Crocopoda

Several recent phylogenetic analyses recovered rhynchosaurs, allokotosaurians (Trilophosaurus, Azendohsaurus, and their kin), prolacertids and archosauriforms as more closely related to each other than to tanystropheids or Protorosaurus speneri (Dilkes, 1998; Modesto & Sues, 2004; Müller, 2004; Gottmann-Quesada & Sander, 2009; Ezcurra, Scheyer & Butler, 2014; Pritchard et al., 2015; Sobral, Sues & Müller, 2015; Nesbitt et al., 2015). This phylogenetic hypothesis is also recovered here and used as the basis to erect the new suprageneric taxon Crocopoda. This clade is here supported by 12 synapomorphies and some of them are features recognized for a long time as apomorphies of a clade nested within Archosauromorpha, such as an ankylothecodont tooth implantation, astragalus with a posterior groove and calcaneum with a prominent tuber. Crocopoda is moderately well supported in this anlaysis, with a Bremer support of 3 (following the pruning of terminals with a large amount of missing data), and moderately stable, with absolute and GC bootstrap frequencies of 50% and 47%, respectively. As defined here, Crocopoda would result in a non-monophyletic group if Protorosaurus speneri and/or Tanystropheus longobardicus are recovered within the less inclusive clade that includes allokotosaurians, rhynchosaurs, and archosauriforms. Therefore, early cladistic analyses and more recently the analysis of Bennett (2012) found a polyphyletic Crocopoda, in which prolacertiforms were more closely related to archosauriforms than to other archosauromorphs, including rhynchosaurs and/or Trilophosaurus. Six additional steps are necessary to recover a non-monophyletic Crocopoda in this data set under a topologically contrained search. In this suboptimal topology, allokotosaurians and rhynchosaurs are successive sister-taxa of tanystropheids and a clade composed of Boreopricea funerea, prolacertids, Tasmaniosaurus triassicus, and archosauriforms. Therefore, the monophyly of Crocopoda is relatively robust based on the results of recent and the current phylogenetic analyses.

The monophyly of Crocopoda implies that the ankylothecodont tooth implantation and a posterior groove on the astragalus appeared only once in non-archosauriform archosauromorphs and they were retained in the common ancestor of archosauriforms. In particular, the ankylothecodont tooth implantation is optimized as having been modified to the thecodont implantation in erythrosuchids and eucrocopod archosauriforms. The presence of a prominent calcaneal tuber seems to have been independently acquired in deeply nested tanystropheids and crocopods (Pritchard et al., 2015).

The monophyly of Allokotosauria and its internal relationships

Nesbitt et al. (2015) recently found a monophyletic group of Triassic non-archosauriform archosauromorphs that is composed of several enigmatic species, namely species of the genus Azendohsaurus and Trilophosaurus, and Pamelaria dolichotrachela, Teraterpeton hrynewichorum, and Spinosuchus caseanus. These authors named this new clade Allokotosauria and recovered it within Crocopoda as the sister-taxon of Prolacerta broomi and archosauriforms. Allokotosauria is also recovered in this analysis, being composed of the three members of the clade included in the current taxonomic sample, namely Pamelaria dolichotrachela, Azendohsaurus madagaskarensis, and Trilophosaurus buettneri. This clade is very well supported here by 21 synapomorphies and with a Bremer support of 12, following the pruning of taxa with a large amount of missing data, and absolute and GC bootstrap frequencies of 93%. Therefore, the result of this analysis strongly supports the phylogenetic hypothesis recovered by Nesbitt et al. (2015). The large number of synapomorphies optimized here for Allokotosauria, in comparison with the five synapomorphies originally found by Nesbitt et al. (2015), could be a result of the more exhaustive character sampling of this analysis but also a consequence of the more restricted taxonomic sample of the clade included here. The inclusion of allokotosaurians with a larger amount of missing data (e.g., Teraterpeton hrynewichorum) may result in ambiguous optimizations for some of the synapomorphies found here for Allokotosauria.

The early Middle Triassic Pamelaria dolichotrachela is recovered here as the most basal member of Allokotosauria, as also found by Nesbitt et al. (2015). However, the support of the monophyly of all allokotosaurians to the exclusion of Pamelaria dolichotrachela was rather poor in Nesbitt et al. (2015), with a Bremer support of 1. In this analysis, the clade composed of Azendohsaurus madagaskarensis and Trilophosaurus buettneri is supported by 15 synapomorphies, but the Bremer support is of 1, and the absolute and GC bootstrap frequencies are of 59% and 24%, respectively. The large difference between both bootstrap frequencies indicates a considerably high amount of conflicting evidence in the resolution of the internal relationships of allokotosaurians. Indeed, only one additional step is necessary to force the placement of Pamelaria dolichotrachela as the sister-taxon of Azendohsaurus madagaskarensis under a topologically constrained search, but to place the former taxon as more closely related to Trilophosaurus buettneri requires 10 additional steps. As a result, on the one hand, the monophyly of Allokotosauria is very robust, but, on the other hand, the position of Pamelaria dolichotrachela is poorly supported and there is evidence that may indicate that it is an azendohsaurid (sensu Nesbitt et al., 2015).

The monophyly and phylogenetic position of Rhynchosauria and its internal relationships

All the supposed rhynchosaurs included in this analysis are recovered in a monophyletic Rhynchosauria (see new definition for the clade in Results). Noteosuchus colletti is found as the oldest member of Rhynchosauria, in agreement with the result of Ezcurra, Scheyer & Butler (2014). However, Ezcurra, Scheyer & Butler (2014) recovered unresolved relationships between Noteosuchus colletti, Mesosuchus browni, and Howesia browni. These relationships are resolved here, in which Noteosuchus colletti is the most basal rhynchosaur, and the early Middle Triassic rhynchosaurs from South Africa (Mesosuchus browni, Howesia browni, and Eohyosaurus wolvaardti) are placed as successive sister-taxa of Rhynchosauridae (Rhynchosaurus articeps and Bentonyx sidensis). This pectinate arrangement of the early Middle Triassic South African rhynchosaurs matches the result of Butler et al. (2015) and Ezcurra, Montefeltro & Butler (2016). Under a topologically constrained search, only one additional step is necessary to find Noteosuchus colletti in multiple positions within Crocopoda and outside Rhynchosauria (e.g., as an allokotosaurian, the most basal crocopod, or an early archosauriform). Similarly, one extra step places Noteosuchus colletti as more closely related to Mesosuchus browni than to other rhynchosaurs, which would be in agreement with the hypothesis of synonymy between both species proposed by Dilkes (1998). Beyond the possible close phylogenetic relationship between both species, Ezcurra, Montefeltro & Butler (2016) show clear features that support the taxonomic distinction of these two taxa. Nevertheless, the monophyly of Rhychosauria is strongly supported in this analysis following the pruning of taxa with a large amount of missing data, with the Bremer support of the group of 16. By contrast, this clade is rather unstable when all their members are included, with absolute and GC bootstrap frequencies of 47% and 38%, respectively.

Rhynchosaurs are recovered here as more closely related to other crocopod archosauromorphs than to allokotosaurians, differing from the results of Pritchard et al. (2015) and Nesbitt et al. (2015), in which rhynchosaurs are the sister-taxon of allokotosaurs and more crownward archosauromorphs. Seven synapomorphies support the clade composed of rhynchosaurs and more crownward archosauromorphs, and three additional steps are necessary to force the position of rhynchosaurs as the most basal crocopods under a topologically constrained search in this analysis. Some authors found that Trilophosaurus was more closely related to rhynchosaurs than to other archosauromorphs (e.g., Gottmann-Quesada & Sander, 2009). Three additional steps are necessary to force this hypothesis (i.e., an allokotosaurian–rhynchosaur clade) in the present phylogenetic analysis. As a result, the monophyly of Rhynchosauria is strongly supported (with the exception of the inclusion of Noteosuchus colletti), but the higher-level relationships of the clade with respect to allokotosaurians are relatively weakly supported and further research seems to be necessary in this part of the archosauromorph tree.

The taxonomic content and monophyly of Proterosuchidae

The 20 terminals sampled in this data matrix that have historically been considered as proterosuchids are found as a polyphyletic assemblage, being placed as tanystropheids (“Exilisuchus tubercularis”), the immediate sister-taxon of archosauriforms (Tasmaniosaurus triassicus), the least crownward archosauriforms (e.g., Proterosuchus spp., Kalisuchus rewanensis, Sarmatosuchus otschevi) and even within Archosauria (“Chasmatosaurus ultimus”) (Fig. 53: blue box). In addition, the proterosuchid clade recovered in the quantitative analysis of Gower & Sennikov (1997) (i.e., Proterosuchus spp., Fugusuchus hejiapanensis and Sarmatosuchus otschevi) is found here as paraphyletic. Fugusuchus hejiapanensis and Sarmatosuchus otschevi are successively more closely related, respectively, to erythrosuchids and eucrocopods than they are to Proterosuchus spp. This result is in agreement with that recovered by Ezcurra, Lecuona & Martinelli (2010). Proterosuchidae sensu Gower & Sennikov (1997) was supported by three synapomorphies that are discussed as follows.

(1) Strongly downturned premaxilla. This character-state (29-2) is restricted to rhynchosaurids (Rhynchosaurus articeps and Bentonyx sidensis), Proterosuchus spp., “Chasmatosaurus” yuani, Archosaurus rossicus, and Sarmatosuchus otschevi among non-archosaurian archosauromorphs (Figs. 16 and 17). The condition present in rhynchosaurids has been clearly acquired independently from that of early archosauriforms. As a result, the distribution of this character-state is consistent with the monophyly of Proterosuchidae sensu Gower & Sennikov (1997). However, this character-state optimizes in the current phylogenetic analysis as the plesiomorphic condition for Archosauriformes, and is subsequently lost in erythrosuchids and eucrocopods.

(2) Fully developed teeth tightly contacting alveolar bone. This condition describes the ankylothecodont tooth implantation (character-state 299-1) and is present in allokotosaurians, rhynchosaurs, Prolacerta broomi, Tasmaniosaurus triassicus, and several early archosauriforms (Fig. 14). As a result, the current phylogenetic analysis optimizes the appearance of ankylothecodont tooth implantation as a synapomorphy of Crocopoda, with the character-state being retained by several non-eucrocopodan archosauriforms (e.g., Proterosuchus spp., Sarmatosuchus otschevi). Therefore, the distribution of this character-state does not support the monophyly of Proterosuchidae sensu Gower & Sennikov (1997).

(3) Cultriform process of the parabasisphenoid dorsoventrally constricted towards the base. The cultriform process of the parabasisphenoid continuously tapers anteriorly, without a dorsoventral constriction of its base, in Proterosuchus fergusi (BP/1/3993) and Proterosuchus goweri (NMQR 880). By contrast, the cultriform process is dorsoventrally compressed at its base in Prolacerta broomi (GHG 431, CT data), Fugusuchus hejiapanensis and Euparkeria capensis (character-state 242-1) (Gower & Sennikov, 1996). The latter condition is optimized as independently acquired by the three above-mentioned species.

Therefore, the only synapomorphy of Proterosuchidae reported by Gower & Sennikov (1997) that is consistent in the current data set with their hypothesis of proterosuchid taxonomic content is the presence of a strongly downturned premaxillary body. However, there are at least eight character-states in this analysis that favour a position of Fugusuchus hejiapanensis and Sarmatosuchus otschevi as more closely related to erythrosuchids and eucrocopods than to Proterosuchus spp. (e.g., 15–22 tooth positions in the maxilla, basal tubera of the basioccipital partially in contact with each other, posterior edge of the intertuberal plate of the parabasisphenoid transversely straight, parabasisphenoid with a posterodorsally-to-anteroventrally oriented main axis). Six additional steps are necessary to recover a monophyletic Proterosuchidae sensu Gower & Sennikov (1997) under a topologically constrained search.

Beyond the taxonomic content of Proterosuchidae hypothesized by the analysis of Gower & Sennikov (1997), there are multiple other taxa that have historically been identified as proterosuchids that are included here in a quantitative analysis for the first time. The majority of these taxa are found to not be members of Proterosuchidae and their phylogenetic positions are discussed in the following categories.

Non-crocopodan archosauromorph. “Exilisuchus tubercularis” from the Early Triassic of Russia is included here for the first time in a quantitative phylogenetic analysis. Ochev (1979) suggested that “Exilisuchus tubercularis” was a probable proterosuchian thecodont and more recently Gower & Sennikov (2000) listed it as part of the Russian proterosuchid record. In this analysis, “Exilisuchus tubercularis” is recovered in an unresolved clade with the tanystropheids Amotosaurus rotfeldensis, Macrocnemus bassanii, and Tanystropheus longobardicus, and this position is supported by the presence of an ilium with a dorsally rimmed caudifemoralis brevis muscle origin on the lateroventral surface of the postacetabular process (character-state 465-1). “Exilisuchus tubercularis” is alternatively placed as the sister-taxon of Tanystropheidae or within this group, and two additional steps are necessary to force the position of this species as a proterosuchid. The low number of extra steps required to modify the position of “Exilisuchus tubercularis” is not surprising because of its large amount of missing data (98.5%).

Immediate sister-taxon of Archosauriformes. Tasmaniosaurus triassicus from the Early Triassic of Australia has historically been considered a proterosuchid (Camp & Banks, 1978; Thulborn, 1986; Ezcurra, Butler & Gower, 2013; Ezcurra, 2014), but is recovered here as the sister-taxon of Archosauriformes (Figs. 48, 49 and 53). Four synapomorphies support the monophyly of Archosauriformes to the exclusion of Tasmaniosaurus triassicus (see synapomorphies of Archosauriformes, above). Three additional steps are necessary to place Tasmaniosaurus triassicus within Proterosuchidae, and one additional step to place it within Archosauriformes, as the sister-taxon of the clade composed of Fugusuchus hejiapanensis and more crownward archosauriforms.

Taxa more crownward than proterosuchids. Several of the putative proterosuchids are recovered here as more closely related to erythrosuchids and eucrocopods than to Proterosuchus spp. Kalisuchus rewanensis, and Vonhuenia friederichi are recovered in a polytomy together with Fugusuchus hejiapanensis, Sarmatosuchus otschevi, Cuyosuchus huenei, and the clade comprising more crownward archosauriforms (Fig. 54). The presence of interdental plates along the entire alveolar margin of the premaxilla, maxilla, and dentary (1:1→2) supports the position of Kalisuchus rewanensis as more crownward among archosauriforms than proterosuchids, and the presence of more than 14 tooth positions in the maxilla excludes it from the clade formed by erythrosuchids and eucrocopods. Vonhuenia friedrichi is found as more crownward than proterosuchids because of the presence of a postzygodiapophyseal lamina in posterior cervical and/or anterior dorsal vertebrae (318:0→1) and cervical and dorsal vertebrae with a gradual transverse expansion of the distal half of the neural spine that lacks mammillary processes (320:2→1). Trees two steps longer include Vonhuenia friedrichi within Proterosuchidae and one additional step forces its position as an erythrosuchid.

Chasmatosuchus rossicus and Chasmatosuchus magnus are found in all the MPTs as more closely related to each other than to other archosauriforms. Chasmatosuchus spp. is recovered as more crownward than proterosuchids and in alternative positions as one of the closest relatives of the clade formed by Sarmatosuchus otschevi, Cuyosuchus huenei, and erythrosuchids (Fig. 60). The position of Chasmatosuchus spp. as more crownward than proterosuchids is supported by the presence of a posterior centrodiapophyseal lamina (316:0→1) and a postzygodiapophyseal lamina in posterior cervical and/or anterior dorsal vertebrae (318:0→1). The monophyly of the genus Chasmatosuchus is supported by the presence of anterior and middle cervical vertebrae with a longitudinal tuberosity strongly developed as a prominent and thick, wing-like shelf extending posteriorly from the base of the transverse process (334:0→1), and postaxial cervical vertebrae with a deep pocket or pit immediately lateral to the base of the neural spine (337:1→2). Two additional steps are required to force both species of Chasmatosuchus to be nested within Proterosuchidae and one further step to place the genus within Erythrosuchidae. When “Gamosaurus lozovskii” and Chasmatosuchus magnus are scored as different terminals (analysis 1), they are recovered in an unresolved clade together with Chasmatosuchus rossicus in all the MPTs (Fig. 60). Therefore, this result is consistent with the hypothesis that “Gamosaurus lozovskii” is a subjective junior synonym of Chasmatosuchus magnus.

Figure 60 First strict reduced consensus tree recovered from analyses 1 indicating the alternatives positions that Prolacertoides jimusarensis, “Ankistrodon indicus,” “Blomosuchus georgii,” SAM-PK-591 and the genus Chasmatosuchus adopt in the most parsimonious trees.

Some clades have been condensed into suprageneric terminals to simplify the figure and their internal relationships are the same as in Figs. 47–51.

The “Long Reef proterosuchid” (SAM P41754; Kear, 2009) is recovered within Erythrosuchidae and may adopt any possible position within the clade. The position of SAM P41754 within Erythrosuchidae is supported by the presence of dorsal vertebrae with a deep pit lateral to the base of the neural spine (361:1→2) and only one additional step is necessary to force the position of this terminal outside the clade. Kear (2009) referred SAM P41754 to Proterosuchidae on the basis of anteroposteriorly elongate centra (length ≥ height/width), elongate neural spines with height > length, and inferred presence of postaxial intercentra. However, these features deserve the following comments. The length of the centrum ranges between 0.93 and 01.23 times the height of its anterior articular facet in the two preserved anterior dorsal vertebrae of SAM P41754, and this range falls within or partially overlaps that observed in the anterior dorsal vertebrae of some erythrosuchids, such as Chalishevia cothurnata (0.95: PIN 4188/98), Garjainia prima (0.80–0.96: PIN 2394/5-14, 5-16) and Garjainia madiba (1.08: BP/1/7135). It should be noted that the strong anteroposterior compression of the presacral vertebrae characteristic of some erythrosuchids and mentioned by Kear (2009) as a difference between this clade and SAM P41754 occurs in the cervico-dorsal transition (character-state 311-1). By contrast, the position of the parapophyses of SAM P41754 indicates that these vertebrae do not belong to the first dorsals, in which the parapophysis is still placed approximately at mid-height in the anterior margin of the centrum (e.g., Garjainia prima: PIN 2394/5), but to more posterior elements in the anterior region of the trunk. Regarding the dorsoventrally tall neural spines of SAM P41754, they also resemble those of erythrosuchids, such as Erythrosuchus africanus (NHMUK PV R3592) and Garjainia prima (PIN 2394/5). Finally, dorsal intercentra are also present in the erythrosuchids Erythrosuchus africanus (Gower, 2003) and Shansisuchus shansisuchus (Wang et al., 2013). The transverse expansion of the distal end of the neural spine of SAM P41754 differs from the gradual expansion of the spine present in proterosuchids (e.g., Proterosuchus alexanderi: NMQR 1484) and some other early archosauriforms (e.g., Vonhuenia friedericki: PIN 1025/11; Sarmatosuchus otschevi: PIN 2865/68), but closely resembles the condition present in the early erythrosuchid Guchengosuchus shiguaiensis (IVPP V8808). Kear (2009) highlighted the common presence of spine tables in SAM P41754 and isolated neural spines referred to the Australian archosauriform Kalisuchus rewanensis (Thulborn, 1979). It is interesting to note that the transverse expansion, with a very rugose distal surface, of these isolated neural spines (which are not considered here unambiguously referable to Kalisuchus rewanensis, see ‘Materials and Methods’) is nearly identical to the condition observed in Guchengosuchus shiguaiensis and may indicate a close phylogenetic relationship between the specimens originally referred to Kalisuchus rewanensis and the Chinese species. In conclusion, there is no unambiguous evidence to refer SAM P41754 to Proterosuchidae and the current results suggest that it is an early erythrosuchid, although the support for this position is very weak.

“Chasmatosaurus ultimus” and Koilamasuchus gonzalezdiazi are recovered deeply nested within Archosauria, as members of Suchia (Fig. 53), which supports the non-proterosuchid affinities for these taxa recently proposed by Ezcurra, Lecuona & Martinelli (2010) and Liu et al. (2015). Koilamasuchus gonzalezdiazi shares with suchians the presence of dorsal vertebrae with a well-rimmed lateral fossa on the centrum below the neurocentral suture (354:1→2), humerus with an approximately symmetric proximal end in anterior view (419:1→0), and preacetabular process of the ilium longer than two-thirds of its height and not extending beyond the level of the anterior margin of the pubic peduncle (460:1→2). In addition, Koilamasuchus gonzalezdiazi is phylogenetically constrained as a suchian more basal than the clade composed of gracilisuchids and paracrocodylomorphs because of the absence of presacral paramedian osteoderms with a distinct longitudinal bend near the lateral edge (598:0→1). Two additional steps are necessary to place Koilamasuchus gonzalezdiazi as the sister-taxon of Euparkeria capensis, which is a more similar position to that recovered by Ezcurra, Lecuona & Martinelli (2010), and six extra steps are required to place it within Proterosuchidae. Regarding “Chasmatosaurus ultimus”, the presence of an antorbital fossa on the ascending and horizontal processes of the maxilla (54:1→2/3) supports its inclusion within Archosauria, and a maxillary alveolar margin on the anterior third of the bone abruptly upturned (70:0→1) and a dentary with a posterocentral process (273:0→1) places this species in a less inclusive clade of Suchia together with gracilisuchids and paracrocodylomorphs. In addition, “Chasmatosuchus ultimus” is found in some MPTs as more closely related to paracrocodylomorphs (e.g., Prestosuchus chiniquensis, Batrachotomus kupferzellensis) than to other pseudosuchians because the palatal process of the maxilla lies distinctly dorsal to the bases of the interdental plates (67:0→1), and the dorsal margin of the anterior portion of the dentary is dorsally expanded compared to the dorsal margin of the posterior portion (271:0→1). The position of “Chasmatosaurus ultimus” outside Proterosuchidae is very well supported, and 11 additional steps are required to place it within Proterosuchidae under a topologically constrained search.

Unambiguous proterosuchids. The results of the present phylogenetic analysis restrict the unambiguous taxonomic content of a monophyletic Proterosuchidae to only five valid species: Archosaurus rossicus, “Chasmatosaurus” yuani, Proterosuchus fergusi, Proterosuchus goweri and Proterosuchus alexanderi. Eleven synapomorphies diagnose Proterosuchidae in the present analysis (see synapomorphies for Proterosuchidae in Results) and its basal branch is very stable (bootstrap frequencies >80%) and well supported (Bremer support of 2 including all terminals and 9 excluding terminals with a large amount of missing data) (Figs. 49 and 57). Twenty additional steps are required to force all taxa historically identified as proterosuchids into a monophyletic group. As a result, the hypothesis of a taxonomically inclusive Proterosuchidae, including all historically referred species, is rejected here. Interestingly, the topologically constrained search to generate a monophyletic group including all taxa historically identified as proterosuchids did not recover a monophyletic Proterosuchia (i.e., Proterosuchidae + Erythrosuchidae).

The interrelationships within Proterosuchidae are completely unresolved when its five unambiguous species are included in the strict consensus tree (Fig. 53). This is a result of the inestability generated by Archosaurus rossicus, which is represented by a single premaxilla, and the topology within the group is resolved in the strict reduced consensus tree generated after the a posteriori pruning of this species. In this strict reduced consensus tree, Proterosuchus goweri is the sister-taxon of “Chasmatosaurus” yuani, and Proterosuchus fergusi and Proterosuchus alexanderi represent the successive sister-taxa, respectively, of this clade (Figs. 48 and 55). The clade including Proterosuchus alexanderi, Proterosuchus goweri and “Chasmatosaurus” yuani to the exclusion of Proterosuchus fergusi is supported by three synapomorphies and that formed by the latter two species is supported by one character-state (see synapomorphy list, above). Two additional steps are necessary to constrain a monophyletic group including only the South African species of the genus Proterosuchus.

The holotype of Proterosuchus fergusi (SAM-PK-591) is recovered in all possible positions within Proterosuchidae or alternatively as one of the closest outgroups of Archosauriformes or sister-taxa of the clade composed of Sarmatosuchus otschevi and more crownward archosauriforms. This result bolsters the conclusion that this specimen cannot be distinguished from other proterosuchid species (Ezcurra & Butler, 2015a).

Ambiguous proterosuchids. “Chasmatosuchus” vjushkovi, “Blomosuchus georgii”, and “Ankistrodon indicus” are recovered as proterosuchids in some of the MPTs (Fig. 60). However, they are also alternatively found immediately outside Archosauriformes or as more closely related to erythrosuchids and eucrocopods than to proterosuchids, forming a massive polytomy with Eorasaurus olsoni, Chasmatosuchus spp., Fugusuchus hejiapanensis, Sarmatosuchus otshevi, Kalisuchus rewanensis, and Cuyosuchus huenei. The ambiguity in the phylogenetic position of these species among proterosuchian-grade archosauromorphs is a result of the lack of preservation of diagnostic characters rather than conflicting phylogenetic signal. Indeed, only one scoring of “Chasmatosuchus” vjushkovi differs from those of Proterosuchus spp. (presence of an anteroposteriorly shallow base of the prenarial process of the premaxilla, character-state 35-0) and scorings are identical for “Ankistrodon indicus,” “Blomosuchus georgii,” and Proterosuchus spp., respectively (=taxonomic equivalents; Wilkinson, 1995). It should be noted that “Chasmatosuchus” vjushkovi is recovered as the sister-taxon of or within the clade composed of Chasmatosuchus magnus and Chasmatosuchus rossicus only in some MPTs and, as a result, the name of the genus of the former species is indicated between inverted commas (Fig. 60). In summary, “Chasmatosuchus” vjushkovi, “Blomosuchus georgii,” and “Ankistrodon indicus” are potential members of Proterosuchidae, but only the former species is currently considered valid (“Blomosuchus georgii” and “Ankistrodon indicus” are regarded here as nomina dubia; see Remarks for both species in ‘Materials and Methods’) and more information is needed to reconstruct unambiguously its phylogenetic relationships.

It is worth mentioning that Vonhuenia friedrichi is not found as more closely related to “Blomosuchus georgii” than to other archosauromorph and one additional step is necessary to force this topology under a topologically constrained search. This result suggests that both species represent different, probably sympatric archosauromorph taxa, in agreement with the hypothesis proposed by Sennikov (1992).

The taxonomic content and monophyly of Erythrosuchidae

The eight species historically identified as erythrosuchids that were included in this analysis (i.e., Garjainia prima, Garjainia madiba, Erythrosuchus africanus, Guchengosuchus shiguaiensis, Shansisuchus shansisuchus, Shansisuchus kuyeheensis, Chalishevia cothurnata, and Uralosaurus magnus) were recovered more closely related to each other than to other archosauromorphs (Fig. 52). Thus, the taxonomic content of Erythrosuchidae in this analysis consists of these species and a specimen from the Early Triassic of Australia that was previously referred to Proterosuchidae (SAM P41754; Kear, 2009; see above). Five synapomorphies diagnose Erythrosuchidae in all the MPTs and two additional synapomorphies are recovered in only some MPTs (see synapomorphies for Erythrosuchidae in Results). The Bremer support of Erythrosuchidae is only 1 and the bootstrap resampling frequencies are also very low (≤5%) when all the terminals are included (Fig. 52), but after the pruning of terminals with a large amount of missing data the Bremer support increases to 18 (Fig. 59). The internal relationships of Erythrosuchidae are represented by a massive polytomy in the strict consensus trees of analyses 1 and 2, but they become almost completely resolved after the pruning of SAM P41754.

Guchengosuchus shiguaiensis is recovered as the sister-taxon of all other erythrosuchids. Erythrosuchus africanus, Shansisuchus shansisuchus, Shansisuchus kuyeheensis, and Chalishevia cothurnata are found in a polytomy together with the clade composed of Garjainia spp. The clade composed of all erythrosuchids to the exclusion of Guchengosuchus shiguaiensis is supported by eight synapomorphies and the group including Erythrosuchus africanus, Shansisuchus shansisuchus, Shansisuchus kuyeheensis and Chalishevia cothurnata is supported by 15 synapomorphies (see synapomorphies of the Shansisuchus shansisuchus + Garjainia prima and Erythrosuchus africanus + Shansisuchus shansisuchus clades, respectively, in Results). The interrelationships between Erythrosuchus africanus, Shansisuchus shansisuchus, Shansisuchus kuyeheensis, and Chalishevia cothurnata are unresolved, but Shansisuchus shansisuchus and Chalishevia cothurnata are recovered as closer to each other than to other erythrosuchids in the strict reduced consensus tree after the a posteriori pruning of Shansisuchus kuyeheensis (Fig. 56). The Shansisuchus shansisuchus and Chalishevia cothurnata clade is supported by two synapomorphies (see synapomorphies of the Shansisuchus shansisuchus + Chalishevia cothurnata clade in Results).

The position of Erythrosuchus africanus as more closely related to Shansisuchus shansisuchus than to Garjainia prima (=“Vjushkovia” triplicostata) is in agreement with the results previously recovered by Parrish (1992) and Gower & Sennikov (1996). By contrast, Ezcurra, Lecuona & Martinelli (2010) found Erythrosuchus africanus as more closely related to Garjainia prima than to Shansisuchus shansisuchus (Fig. 5D). Ten additional steps are necessary to force a position for Erythrosuchus africanus as more closely related to Garjainia spp. than to Shansisuchus spp. and Chalishevia cothurnata.

Parrish (1992) reported two synapomorphies supporting the position of Erythrosuchus africanus as more closely related to Shansisuchus shansisuchus than to Garjainia prima, which are discussed as follows in the context of the present phylogenetic analysis.

(1) The external surface of the maxilla adjacent to the anterior and anterodorsal surfaces of the antorbital fenestra is recessed. Parrish (1992) described Garjainia prima as lacking an antorbital fossa on the maxilla, resembling the condition in Guchengosuchus shiguaiensis, Fugusuchus hejiapanensis, and proterosuchids (Fig. 17). However, Garjainia prima possesses an extensive antorbital fossa on the ascending process of the maxilla, although this fossa does not reach the base of the process as in Erythrosuchus africanus, Chalishevia cothurnata, and Shansisuchus shansisuchus (Figs. 19 and 22). As a result, although the observation of Parrish (1992) regarding Garjainia prima was incorrect, the presence of an antorbital fossa reaching the base of the ascending process and extending on the horizontal process of the maxilla (character-state 54-2) is a feature that supports the hypothesis that Erythrosuchus africanus is more closely related to Shansisuchus shansisuchus than to Garjainia prima.

(2) Absence of a pineal foramen. Gower (2003) reported that there is no conclusive evidence for the presence or absence of a pineal foramen in available specimens of Erythrosuchus africanus. This is followed here for the vast majority of specimens referred to Erythrosuchus africanus, but in NMQR 1473 this area of the skull is well preserved and there is no pineal foramen (character-state 164-2) (Fig. 23A), resembling the condition in Shansisuchus shansisuchus (IVPP V2501, 2503, 2506, 2508). The holotype of Garjainia prima possesses a small, circular pit that is slightly displaced to the left of the median line of the skull (PIN 2394/5). This pit may represent a vestigial pineal foramen, but this interpretation should be considered tentative and it is scored here as a question mark. Beyond the presence or absence of a pineal foramen in Garjainia prima, this feature is absent in the most immediate outgroups of the clade composed of Garjainia spp., Shansisuchus spp., and Erythrosuchus (e.g., Fugusuchus hejiapanensis: Cheng, 1980; Guchengosuchus shiguaiensis: IVPP V8808-2; Euparkeria capensis: SAM-PK-5867). As a result, if a pineal foramen is present in Garjainia prima it may represent an autapomorphy of this taxon rather than a symplesiomorphy of Erythrosuchidae that is apomorphically lost in Erythrosuchus africanus and Shansisuchus shansisuchus. The absence or presence of a pineal foramen cannot be determined in Shansisuchus kuyeheensis and Chalishevia cothurnata because this region of the skull is not preserved.

Gower & Sennikov (1996) proposed two synapomorphies supporting Erythrosuchus africanus being more closely related to Shansisuchus shansisuchus than to Garjainia prima. These two character-states are discussed as follows.

(1) Ventral ramus of the opisthotic recessed. Gower & Sennikov (1996) described the ventral ramus of the opisthotic as poorly developed and recessed within the stapedial groove in Erythrosuchus africanus and Shansisuchus shansisuchus (character-state 218-1). By contrast, in Garjainia prima, Garjainia madiba, Fugusuchus hejiapanensis, Sarmatosuchus otschevi, and proterosuchids the ventral ramus of the opisthotic is better developed and well exposed in lateral view (Gower & Sennikov, 1996; Gower et al., 2014). The condition of this character cannot be determined in Shansisuchus kuyeheensis and Chalishevia cothurnata. As a result, the poor development of the ventral ramus of the opisthotic supports the hypothesis of Gower & Sennikov (1996).

(2) Medial margins of exoccipitals make contact for majority of their length. The exoccipitals contact each other on the floor of the endocranial cavity but diverge posteriorly on the dorsal surface of the occipital condyle in proterosuchids (e.g., Proterosuchus goweri: NMQR 880; Proterosuchus alexanderi: NMQR 1484; Proterosuchus fergusi: SAM-PK-K10603), Fugusuchus hejiapanensis (Gower & Sennikov, 1996), Sarmatosuchus otschevi (PIN 2865/68-1), and Garjainia prima (PIN 951/60). By contrast, the exoccipitals form the entire floor of the posterior end of the endocranial cavity and meet along the dorsal surface of the occipital condyle in Erythrosuchus africanus and Shansisuchus shansisuchus (Gower & Sennikov, 1996; Gower, 1997) (character-state 221-2). The condition of this character cannot be determined in Shansisuchus kuyeheensis and Chalishevia cothurnata. Therefore, this condition also supports the position of Erythrosuchus africanus as more closely related to Shansisuchus shansisuchus than to Garjainia prima.

Ezcurra, Lecuona & Martinelli (2010) found the following two character-states that supported the position of Erythrosuchus africanus as more closely related to Garjainia prima than to Shansisuchus shansisuchus:

(1) Height of middle dorsal neural spines equal to or more than 50% of the total height of the vertebra. The neural spines of the middle dorsal vertebrae of Shansisuchus shansisuchus are proportionally dorsoventrally tall, but they represent slightly less than half of the total height of the vertebra (ca. 44%–46%; Young, 1963: Figs. 21D and 21F). In Garjainia prima, the neural spines represent approximately half of the total length of the vertebra (Huene, 1960). The condition of this character cannot be determined in Shansisuchus kuyeheensis and Chalishevia cothurnata. Although the neural spines of the middle dorsal vertebrae of Garjainia prima seems to be slightly taller than those of Shansisuchus shansisuchus, their difference in ratio is not enough (i.e., <5%) to support a phylognenetic distinction based on this feature.

(2) Ischial length more than twice the anteroposterior length of the acetabulum. The total length of the ischium is approximately 2.5 times the length of the acetabulum in Garjainia prima (PIN specimen: 2.43) and Garjainia madiba (BP/1/5525: 2.40), whereas this ratio is lower in Erythrosuchus africanus (SAM-PK-905: 1.90) and Shansisuchus shansisuchus (Young, 1964: Fig. 30B: 1.90). However, the cluster analysis of the ratios of this character groups all the erythrosuchids within the same cluster (character-state 482-1). For example, other archosauromorphs closely related to erythrosuchids in which this character can be scored possess ratios that bridge the gap between erythrosuchids, such as Prolacerta broomi (BP/1/2676: 1.92) and Cuyosuchus huenei (MCNAM 2669: 2.19). As a result, the ratios calculated for erythrosuchids do not support a Garjainia spp. and Erythrosuchus africanus clade, excluding Shansisuchus shansisuchus, in this data matrix.

The discussion of the character-states previously found to be phylogenetically informative in the relationships between Garjainia prima, Erythrosuchus africanus and Shansisuchus shansisuchus clearly indicates that there is more evidence favouring an Erythrosuchus africanus + Shansisuchus shansisuchus clade to the exclusion of Garjainia prima. In agreement, the Erythrosuchus africanus and Shansisuchus shansisuchus clade is very well supported in this analysis, with a Bremer support of 9 following the pruning of terminals with a large amount of missing data and 10 additional steps are necessary to force a Garjainia prima + Erythrosuchus africanus clade to the exclusion of Shansisuchus shansisuchus under a topologically constrained search.

GHG 7433MI is a probable juvenile erythrosuchid from the Cynognathus AZ of South Africa that was tentatively referred to Erythrosuchus africanus by Gower (2003). The results of the present phylogenetic analysis do not support this assignment and, instead, found this specimen more closely related to Garjainia spp. than to other erythrosuchids. A detailed anatomical study of this specimen is necessary in order to determine its taxonomic affinities confidently, but the presence of a horizontal process of the maxilla with a dorsoventrally expanded posterior end that possesses a straight ventral margin of the antorbital fenestra supports its probable referral to the genus Garjainia (character-state 63-2). Two additional steps are necessary to force a sister-taxon relationship between GHG 7433MI and Erythrosuchus africanus. The monophyly of the genus Garjainia is very well supported in this analysis after the pruning of terminals with a large amount of missing data, with a Bremer support value of 8 (Fig. 59). As a result, GHG 7433MI is tentatively referred to Garjainia sp.

Non-erythrosuchids previously considered as members of the group. Cuyosuchus huenei from the Late Triassic of Argentina and “Dongusia colorata” from the early Middle Triassic of Russia have been referred to Erythrosuchidae by some previous authors (e.g., Tatarinov, 1961; Hughes, 1963; Ewer, 1965; Romer, 1966; Romer, 1972a; Cruickshank, 1972). However, Cuyosuchus huenei was subsequently reinterpreted as more closely related to crown archosaurs than to erythrosuchids (i.e., a eucrocopod archosauriform) by Desojo, Arcucci & Marsicano (2002), and “Dongusia colorata” as a probable rauisuchian archosaur by Gower & Sennikov (2000). The result of this phylogenetic analysis agrees with the interpretations that these two species do not belong to Erythrosuchidae. Cuyosuchus huenei is recovered as the sister-taxon of the clade that includes erythrosuchids and eucrocopods, contrasting with the interpretation of Desojo, Arcucci & Marsicano (2002) that this species was a eucrocopod. Two synapomorphies exclude Cuyosuchus huenei from the less inclusive clade that includes erythrosuchids and eucrocopods (see synapomorphies of the Erythrosuchidae + Archosauria clade in Results) and only one additional step is necessary to include this species within Erythrosuchidae or at the base of Eucrocopoda. As a result, the phylogenetic position of Cuyosuchus huenei is rather weakly supported.

“Dongusia colorata” is found as a paracrocodylomorph pseudosuchian, which is consistent with the hypothesis of Gower & Sennikov (2000) that the species may belong to a rauisuchian. This species is included within Suchia based on the presence of dorsal vertebrae with a well-rimed lateral fossa on the centrum below the neurocentral suture (character-state 354:1→2) and among paracrocodylomorphs based on the presence of middle and posterior dorsal vertebrae with a hyposphene-hypantrum accessory intervertebral articulation (character-state 359:0→1). Two additional steps are necessary to force the position of “Dongusia colorata” outside Pseudosuchia and as an erythrosuchid under topologically constrained searches. The low number of additional steps necessary to modify the relationships of this taxon is not unexpected because of the high proportion of missing data present in its scorings (97.3%).

The monophyly of Eucrocopoda

A close phylogenetic relationship between Euparkeria capensis, proterochampsids, doswelliids, and archosaurs to the exclusion of proterosuchids and erythrosuchids has been repeatedly recovered by the vast majority of recent phylogenetic analyses including the present one (e.g., Gower & Sennikov, 1997; Ezcurra, Lecuona & Martinelli, 2010; Desojo, Ezcurra & Schultz, 2011; Nesbitt, 2011; Bennett, 2012). This topology has been used here as the basis to erect the new archosauriform clade Eucrocopoda (see Results). This clade is diagnosed by four unambiguous apomorphies, but the branch is very well supported following the a posteriori pruning of terminals with a large amount of missing data, with a Bremer support of 12. Because of the definition proposed here for Eucropoda, this clade would be non-monophyletic if Proterosuchus fergusi and/or Erythrosuchus africanus are recovered within the less inclusive clade that includes Euparkeria capensis, Proterochampa barrionuevoi, Doswellia kaltenbachi, and archosaurs. As a result, a non-monophyletic Eucrocopoda has been recovered only by Dilkes & Sues (2009) among recent quantitative phylogenetic analyses, in whose analysis Erythrosuchus africanus was more closely related to more crownward archosauriforms to the exclusion of Euparkeria capensis. However, modified versions of the data matrix of Dilkes & Sues (2009) recovered a monophyletic Eucrocopoda (e.g., Ezcurra, Lecuona & Martinelli, 2010; Desojo, Ezcurra & Schultz, 2011) and 14 additional steps are necessary here to recover the topology found by Dilkes & Sues (2009). Therefore, there is very strong evidence supporting the monophyly of Eucrocopoda. The monophyly of Eucrcopoda implies that femoral character-states historically recognized as apomorphies of a less inclusive clade of archosauriforms appeared once in the evolution of the clade, including the presence of an anteromedially oriented femoral head and a fourth trochanter.

The phylogenetic positions of Dongusuchus and Yarasuchus

Dongusuchus efremovi and Yarasuchus deccanensis are two enigmatic archosauriforms from the early Middle Triassic of Russia and India, respectively (see ‘Materials and Methods’). These two species were recovered here as more closely related to each other than to other archosauriforms and within Eucrocopoda, in a trichotomy together with Euparkeria capensis and the clade composed of proterochampsians and archosaurs. The clade Dongusuchus efremovi + Yarasuchus deccanensis is supported by two synapomorphies (see synapomorphies of the clade Dongusuchus efremovi + Yarasuchus deccanensis in Results) and two additional steps are necessary to break up its monophyly. The position of the Dongusuchus efremovi + Yarasuchus deccanensis clade found here is consistent with the topology recovered by Niedźwiedzki, Sennikov & Brusatte (2014), which places Dongusuchus efremovi in a polytomy with other non-archosaurian eucrocopods (Yarasuchus deccanensis was not included in their analysis).

Beyond the two synapomorphies found for this group, the very similar overall morphology between the femora of Dongusuchus efremovi and Yarasuchus deccanensis suggests that these two species are closely related. Indeed, the scorings of both species are completely consistent between each other in the data matrix. The close phylogenetic relationship and overall morphological similarity between these species suggest biogeographical affinities between the areas that are now India and Russia during the early Middle Triassic. These two areas were broadly palaeolatitudinally separated from each other at that time and this evidence is congruent with the hypothesis of global cosmopolitanism among Middle Triassic tetrapods (Ezcurra, 2010b).

It is also worth mentioning that among the specimens equivocally referred to Dongusuchus efremovi, Niedźwiedzki, Sennikov & Brusatte (2014) described the presence of an autapomorphic helical groove along the posterior edge of the medial surface on the proximal portion of the ulna. This feature is absent on the preserved ulnae of Yarasuchus deccanensis (ISI R334) and, as a result, the phylogenetic result of this analysis does not shed light on the assignment of the equivocally referred bones of Dongusuchus efremovi.

The phylogenetic positions of Euparkeria and Proterochampsidae

Euparkeria capensis and the proterochampsids have been repeatedly hypothesized as being among the closest relatives of Archosauria since the first quantitative phylogenetic analyses (e.g., Benton & Clark, 1988; Sereno & Arcucci, 1990). One exception to this general consensus is the result recovered by Dilkes & Sues (2009), which placed Euparkeria capensis as the sister-taxon of Erythrosuchus africanus and more crownward archosauriforms. However, this latter result has not been repeated in more recent phylogenetic studies (e.g., Brusatte et al., 2010; Ezcurra, Lecuona & Martinelli, 2010; Desojo, Ezcurra & Schultz, 2011; Nesbitt, 2011; Dilkes & Arcucci, 2012). By contrast, the relative phylogenetic positions of Euparkeria capensis and proterochampsids with respect to Archosauria are much more debated. Proterochampsids have been found as the sister-taxon of Archosauria and Euparkeria capensis as a less crownward archosauriform by multiple, independent phylogenetic analyses in the last 30 years (e.g., Sereno, 1991; Parrish, 1993; Juul, 1994; Benton, 2004; Ezcurra, Lecuona & Martinelli, 2010; Desojo, Ezcurra & Schultz, 2011; Schoch & Sues, 2014). Nevertheless, Benton & Clark (1988) found Euparkeria capensis as the sister-taxon of Archosauria and proterochampsids as less crownward archosauriforms, and the same result was more recently recovered by Nesbitt (2011), Dilkes & Arcucci (2012), Bennett (2012), Sookias et al. (2014a) and Sookias et al. (2014b) (but phytosaurs were unambiguously found outside Archosauria in these analyses, with the exception of Bennett (2012)).

The present phylogenetic analysis recovered Euparkeria capensis as the sister-taxon of the clade comprising proterochampsids and archosaurs (Figs. 50, 51 and 56), resembling the topology most commonly recovered in previous studies. A total of six synapomorphies support this hypothesis (see synapomorphies for the Proterochampsia + Archosauria clade in Results) and three of them were also sampled in the analysis of Nesbitt (2011), which placed Euparkeria capensis as the closest sister-taxon of Archosauria. The scorings for these three characters are compared between the two data matrices as follows.

(1) Postparietal absent as a separate ossification (character-state 171-2). This character-state is included under character 146 of Nesbitt (2011). The scorings of this character by Nesbitt (2011) are congruent with those of the current study, with the exception that Mesosuchus browni is scored here as lacking a postparietal, in agreement with the scoring of Dilkes (1998: character 29). Tasmaniosaurus triassicus, proterosuchids, erythrosuchids, Euparkeria capensis and probably Asperoris mnyama possess a postparietal, which is absent in proterochampsids, doswelliids, and archosaurs (Ewer, 1965; Cruickshank, 1972; Gower, 2003; Dilkes & Sues, 2009; Nesbitt, 2011; Nesbitt, Butler & Gower, 2013; Ezcurra, 2014). As a result, the scorings for this character support the hypothesis of Euparkeria capensis as lying outside the Proterochampsidae + Archosauria clade.

(2) Parabasisphenoid horizontal (character-state 235-0). This character-state is included in character 97 of Nesbitt (2011), in which all the archosauromorphs sampled were scored as possessing a more vertical parabasisphenoid, with the exception of Mesosuchus browni, Prolacerta broomi, and Proterosuchus fergusi (Figs. 27 and 28). The scorings for this character in the present analysis are generally in agreement with those of Nesbitt (2011). However, the bases of the basal tubera and the basipteryoid processes are at approximately the same level in lateral view in the parabasisphenoid of proterochampsids (e.g., Chanaresuchus bonapartei: MCZ 4037, PULR 07, PVL 4586; Tropidosuchus romeri: PVL 4601, 4606; Proterochampsa barrionuevoi: PVL 2063), Doswellia kaltenbachi (USNM 214823), phytosaurs (Parasuchus hislopi: ISI R42; Parasuchus angustifrons: BSPG 1931 X 502; Nicrosaurus kapffi: NHMUK PV R42743), Riojasuchus tenuisceps (PVL 3827) and ornithodirans (e.g., Marasuchus lilloensis: PVL 3872; Lewisuchus admixtus: PULR 01; Silesaurus opolensis: Dzik, 2003; Herrerasaurus ischigualastensis: PVSJ 407) and these taxa, as a result, are scored as possessing a horizontal parabasisphenoid. The present phylogenetic analysis optimizes an oblique (=more vertical) parabasisphenoid as symplesiomorphic for erythrosuchids, Dorosuchus neoetus, and Euparkeria capensis, whereas a horizontal parabasisphenoid is recovered as a synapomorphy of the clade that includes proterochampsians and archosaurs. An oblique parabasisphenoid is secondarily acquired by the doswelliid Archeopelta arborensis and suchian archosaurs.

(3) Absence of postaxial cervical intercentra (character-state 346-1). Nesbitt (2011) scored postaxial intercentra (character 177 of his data matrix) as present in Mesosuchus browni, Prolacerta broomi, Proterosuchus fergusi, Erythrosuchus africanus, and Euparkeria capensis, and absent in proterochampsids, Vancleavea campi, and archosaurs. The scorings in Nesbitt (2011) are in agreement with those of the present data matrix and support a less crownward position for Euparkeria capensis than proterochampsids and archosaurs.

Nesbitt (2011) found seven unambiguous synapomorphies in support of the hypothesis that Euparkeria capensis is more closely related to Archosauria than to other non-archosaurian archosauriforms. All of these characters are included in the current analysis and are discussed as follows.

(1) Foramen on the medial side of the articular. Nesbitt (2011: character 159) scored a foramen for the passage of the chorda tympani on the medial side of the articular as present in Euparkeria capensis, phytosaurs, Riojasuchus tenuisceps and most suchians. However, this foramen is recognized here as more broadly distributed among non-archosaurian archosauriforms, being present in the proterosuchids Proterosuchus fergusi (RC 846) and Proterosuchus alexanderi (NMQR 1484), and the erythrosuchids Garjainia madiba (NMQR 3051) and Erythrosuchus africanus (Gower, 2003: 35) (character-state 294-1). As a result, the presence of a medial foramen in the articular is optimized here as symplesiomorphic for Archosauriformes, being retained by eucrocopods, and subsequently lost in proterochampsians and ornithodirans.

(2) Distal ends of neural spines of the cervical vertebrae laterally expanded, and (3) neural spines of the dorsal vertebrae with a lateral expansion and a flat dorsal margin. See discussion about the distribution of the presence of a gradual transverse expansion of the neural spine in the presacral vertebrae in character 320, and the presence of a spine table is scored as a different character in this phylogenetic analaysis (character 321; see discussions of both characters for the logical basis of their separation). A gradual transverse expansion of the neural spine is optimized here as a symplesiomorphy of Archosauriformes, which is apomorphically lost in proterochampsids and archosaurs. On the other hand, the presence of spine tables in cervical and dorsal vertebrae is recovered as an apomorphy of Pseudosuchia.

(4) Proximal end of the fibula, in proximal view, rounded or slightly elliptical. Nesbitt (2011: character 341) recognized the presence of a rounded to slightly elliptical fibula in proximal view in Euparkeria capensis, phytosaurs, ornithosuchids and several non-crocodylomorph suchians. By contrast, proterosuchids, erythrosuchids, and proterochampsids were scored as having a transversely compressed proximal end of the fibula. A more variable distribution of this character among archosaurs is recognized here, in which a transversely compressed proximal end of the fibula (character-state 525-1) is present in Nicrosaurus kapffi (Huene, 1923: Fig. 54), Turfanosuchus dabanensis (IVPP V3237) and Aetosauroides scagliai (PVL 2073). The presence of a transversely compressed fibula in proximal view is optimized here as symplesiomorphic of the clade that includes tanystropheids and crocopods and it is retained in ornithodirans and the common ancestor of pseudosuchians. As a result, the presence of a rounded or slightly elliptical fibula in proximal view is optimized as autapomorphic for Euparkeria capensis.

(5) Distal end of the fibula asymmetrical in lateral view. The presence of an asymmetric distal end of the fibula in lateral view was recognized by Nesbitt (2011: character 345) to be present in Euparkeria capensis, phytosaurs, ornithosuchids and non-poposauroid basal suchians. The scorings for this character in the present phylogenetic analysis (character 531) are generally similar to those of Nesbitt (2011), but the inclusion of phytosaurs within Pseudosuchia produces an ambiguity in its optimization at the base of Archosauria because of the variable distribution of the feature among the most basal ornithodirans (e.g., a symmetric distal end of tibia in Lagerpeton chanerensis and an asymmetric distal end in Marasuchus lilloensis). The ambiguity in the optimization of this character is extended up to the base of Eucrocopoda.

(6) The posterior corner of the dorsolateral margin of the astragalus dorsally overlaps the calcaneum much more than the anterior portion. Nesbitt (2011: character 360) scored this character-state as present in Euparkeria capensis and all the archosaurs sampled in his analysis. However, this character-state is recognized here to be present only in the ornithosuchid Riojasuchus tenuisceps (PVL 3827), whereas in other archosauromorphs the dorsolateral margin of the astragalus equally overlaps the anterior and posterior portions of the calcaneum (character-state 541-0). Therefore, this character is not phylogenetically informative in the present analysis.

(7) Calcaneal tuber shaft proportions about the same as or broader than tall. The scorings for the proportions of the shaft of the calcaneal tuber are almost entirely congruent in this and Nesbitt’s (2011) data sets. The only exception is that Marasuchus lilloensis is scored here as having a taller than broad calcaneal tuber (PVL 3870, character-state 547-0) rather than approximately as tall as broad (Nesbitt, 2011: character-state 376-1). Euparkeria capensis possesses a calcaneal tuber with a shaft that is approximately as tall as broad, resembling the condition in non-suchian pseudosuchians (Nesbitt, 2011) and some basal ornithodirans (e.g., Asilisaurus kongwe: Nesbitt et al., 2010). As a result, this character agrees with the hypothesis of Euparkeria capensis rather than proterochampsids being more closely related to archosaurs.

The present revision of the currently available evidence in support of the position of either proterochampsids or Euparkeria capensis as the sister-taxon of Archosauria clearly favours the hypothesis of a more basal position for Euparkeria capensis with respect to proterochampsids. Indeed, only one character is recognized here as providing unambiguous support for the alternative hypothesis. The more crownward position of proterochampsians with respect to other non-archosaurian archosauriforms is moderately well supported, with a Bremer support value of 4 (Fig. 59, after the exclusion of taxa with a large amount of missing data) and three additional steps are required to force the monophyly of Euparkeria capensis + Archosauria to the exclusion of proterochampsids under a topologically constrained search.

It should be also noted that Dorosuchus neoetus is found in all the MPTs as a less crownward archosauriform than Euparkeria capensis, contrasting with the topology reported by Sookias et al. (2014a), which has found Dorosuchus neoetus as the immediate sister-taxon of Phytosauria + Archosauria. Nevertheless, the result recovered here agrees with that of Sookias et al. (2014a) in rejecting the monophyly of Euparkeria capensis and Dorosuchus neoetus within a taxonomically inclusive Euparkeriidae (see discussion about the taxonomic content of the group in Sookias & Butler, 2013; the taxonomic content of the monophyletic Euparkeriidae found by Sookias et al. (2014b) was not tested in this analysis). The different results of the phylogenetic analysis of Sookias et al. (2014a) and those reported here are mainly because of the decision taken here to exclude the tentatively referred specimens of Dorosuchus neoetus from this terminal rather than differences in character scorings. The more crownward position of Euparkeria capensis than Dorosuchus neoetus is supported by two braincase synapomorphies (see synapomorphies of the clade Euparkeria capensis + Archosauria in Results) and the Bremer support is 2 and bootstrap frequencies are relatively low (<50%). Two additional steps are required to recover Euparkeria capensis as less crownward than Dorosuchus neoetus or to force the monophyly of Euparkeriidae under topologically constrained searches.

The phylogenetic position of Doswelliidae

Doswellia kaltenbachi is a heavily armoured and probably semi-aquatic archosauriform from the Late Triassic of North America (Weems, 1980; Dilkes & Sues, 2009; Sues, Desojo & Ezcurra, 2013). The phylogenetic position of this species has been matter of debate with little consensus since its original description by Weems (1980) (Sues, Desojo & Ezcurra, 2013). The family Doswelliidae was monospecific for nearly 30 years following the description of its type genus. A monophyletic group of doswelliid species has been recognized only recently (Desojo, Ezcurra & Schultz, 2011) and currently includes six species from the Middle and Upper Triassic of South America, North America and Europe (Heckert, Lucas & Spielmann, 2012; Lucas, Spielmann & Hunt, 2013; Sues, Desojo & Ezcurra, 2013; Schoch & Sues, 2014). All the quantitative phylogenetic analyses that have tested the position of Doswellia kaltenbachi or doswelliids as a whole have agreed in their placement within Eucrocopoda, but lying outside the crown-group Archosauria (Benton & Clark, 1988; Dilkes & Sues, 2009; Ezcurra, Lecuona & Martinelli, 2010; Desojo, Ezcurra & Schultz, 2011; Schoch & Sues, 2014). Doswelliids have been alternatively recovered as more closely related to archosaurs than to other archosauriforms (Desojo, Ezcurra & Schultz, 2011) or forming a clade with the also probably semi aquatic proterochampsids and/or the fully aquatic Vancleavea campi (Benton & Clark, 1988; Dilkes & Sues, 2009; Ezcurra, Lecuona & Martinelli, 2010; Schoch & Sues, 2014).

The present analysis recovered all the sampled supposed doswelliids as a monophyletic group more closely related to Vancleavea campi and proterochampsids than to other archosauriforms (Figs. 50 and 51). This result agrees partially with those found by some previous studies, but particularly matches the phylogenetic hypothesis proposed recently by Schoch & Sues (2014), in which Vancleavea campi is the sister-taxon to all other doswelliids, and proterochampsids are their immediate sister-taxon. As a result, the families Proterochampsidae and Doswelliidae are included within the clade Proterochampsia because of its definition (Nesbitt, 2011).

Ezcurra, Lecuona & Martinelli (2010) found two synapomorphies supporting the position of doswelliids (Doswellia kaltenbachi and Vancleavea campi) as more closely related to archosaurs than to Chanaresuchus bonapartei. These two character-states are discussed as follows.

(1) Occipital condyle at the same level as the craniomandibular joint. The present phylogenetic analysis optimizes the presence of an occipital condyle anterior to the level of the craniomandibular joint (character-state 228-1) as plesiomorphic for archosauromorphs. As a result, an occipital condyle at the same level as the craniomandibular joint is interpreted as independently and apomorphically acquired in Doswellia kaltenbachi (USNM 214823) and Smilosuchus spp. (UCMP 27200) (Fig. 26), and does not provide evidence for the non-monophyly of Proterochampsia.

(2) Absence of a posterior groove on the astragalus. The posterior groove of the astragalus is present in all the archosauriforms sampled in the present analysis, with the exception of most ornithodirans, Vancleavea campi, and Tropidosuchus romeri (character-state 539-1). Therefore, the posterior groove of the astragalus is optimized as independently lost in these three latter taxa, and this character-state is congruent with the monophyly of Proterochampsia.

Desojo, Ezcurra & Schultz (2011) recovered four synapomorphies supporting a closer relationship between doswelliids and more crownward archosauriforms than with Chanaresuchus bonapartei. These four character-states are discussed as follows.

(1) Ventral process of the postorbital ends close to or at the ventral margin of the orbit. This character-state cannot be evaluated in Doswellia kaltenbachi because this species lacks a distinct ventral process of the postorbital. As a result, the character is considered here inapplicable for Doswellia kaltenbachi and the condition is unknown in Tarjadia ruthae, Archeopelta arborensis, and Jaxtasuchus salomoni. Vancleavea campi possesses a ventral process of the postorbital that ends well above the ventral margin of the orbit (Nesbitt et al., 2009), resembling the condition in the vast majority of non-archosaurian archosauriforms. Therefore, the extension of the ventral process of the postorbital does not support the non-monophyly of Proterochampsia.

(2) Ventral process of the squamosal anteroventrally projected and constricting the infratemporal fenestra at mid-height. This character-state is restricted to phytosaurs, gracilisuchids, and the dinosaur Heterodontosaurus tucki among the taxa sampled in the present phylogenetic analysis. By contrast, Doswellia kaltenbachi (USNM 214823) possesses a vertical ventral process of the squamosal, which represents the symplesiomorphic condition for Archosauriformes.

(3) Posterior process of the squamosal ventrally curved. Doswellia kaltenbachi retains the symplesiomorphic condition for Archosauriformes of a straight posterior process of the squamosal (USNM 214823). In the present analysis, the presence of a ventrally curved posterior process of the squamosal is found as an independently derived state of both Pseudosuchia and some rhadinosuchines.

(4) Absence of a semilunar depression on the parabasisphenoid. The condition of this character could not be determined for any doswelliid based on first-hand observations of all the available specimens of the species sampled in the present phylogenetic analysis.

In summary, there is no strong evidence for the non-monophyly of Proterochampsia. Conversely, 13 synapomorphies support the monophyly of Proterochampsia (see synapomorphies of Proterochampsia in Results) and the clade has a Bremer support value of 10 when terminals with a large amount of missing data are pruned a posteriori (Fig. 59). Thirteen additional steps are requiered to force Doswellia kaltenbachi to be more closely related to archosaurs than to proterochampsids, and the monophyly of Doswelliidae is still recovered in this topologically constrained search. The monophyly of Doswelliidae is supported by nine synapomorphies (see synapomorphies of Doswelliidae in Results) and the Bremer support for the group is 5 following the pruning of terminals with a large amount of missing data (i.e., Doswelliidae being only composed of Doswellia kaltenbachi and Vancleavea campi). Five additional steps are necessary to place Vancleavea campi as the sister-taxon of all other proterochampsians, and ten additional steps to obtain the phylogenetic hypothesis recovered by Nesbitt (2011), in which Vancleavea campi is less crownward than Euparkeria capensis, proterochampsids and archosaurs. As a result, the hypothesis that doswelliids are more closely related to archosaurs than to other archosauriforms is rejected here, as well as the hypothesis that Vancleavea campi is a less crownward archosauriform than Euparkeria capensis, proterochampsids and archosaurs. The observations and results of the present phylogenetic analysis strongly suggest the presence of a large clade of semi aquatic to aquatic non-archosaurian eucrocopods composed of proterochampsids and doswelliids.

The phylogenetic position of Phytosauria

The vast majority of quantitative phylogenetic analyses of the last 30 years consistently recovered phytosaurs as basal pseudosuchians, and in many cases as the sister-taxon of all other members of this clade (e.g., Gauthier, 1984; Benton & Clark, 1988; Sereno & Arcucci, 1990; Sereno, 1991; Benton, 1999; Nesbitt & Norell, 2006; Nesbitt, 2007; Brusatte et al., 2010). However, Nesbitt (2011) found phytosaurs as the immediate sister-taxon of Archosauria, and this result has been subsequent recovered by other studies that employed modified versions of this data set (e.g., Dilkes & Arcucci, 2012; Nesbitt & Butler, 2013; Baczko, Desojo & Pol, 2014; Nesbitt et al., 2014; Sookias et al., 2014a; Sookias et al., 2014b). The phylogenetic position of phytosaurs outside Archosauria has important evolutionary implications, such as for the origin of the crurotarsal ankle joint and the morphological disparity that the crown group achieved during the Triassic (Nesbitt, 2011). In addition, the optimization of multiple characters at the base of Archosauria and its most immediate sister nodes is directly dependant on the position of phytosaurs.

In the present phylogenetic analysis, phytosaurs are within Archosauria and as the sister-group of all other pseudosuchians (=Crurotarsi sensu Sereno & Arcucci (1990)) (Figs. 50–52). The phylogenetic position of phytosaurs within Pseudosuchia is supported by 13 synapomorphies in this analysis (see synapomorphies of Pseudosuchia in Results) and six additional steps are necessary to force the placement of phytosaurs as the sister-taxon of Archosauria under a topologically constrained search. The Bremer support for Pseudosuchia is of 6, following the a posteriori pruning of terminals with a large amount of missing data, and the absolute and GC bootstrap frequencies are of 33% and 23%, respectively (Fig. 59). Nesbitt (2011) found 10 synapomorphies supporting the monophyly of archosaurs to the exclusion of phytosaurs, and these character-states are discussed as follows.

(1) Palatal processes of the maxillae meet at the midline. The scorings of this character are congruent in both data matrices, with ornithodirans, ornithosuchids, and suchians possessing contact between the palatal processes of the maxillae along the midline (character-state 32-1 of Nesbitt (2011), character-state 66-2 of the present analysis). By contrast, in phytosaurs and less crownward archosauriforms the palatal processes are less medially developed and lack such a contact (Fig. 26). The distribution of this character-state is thus consistent with the hypothesis that phytosaurs lie outside Archosauria.

(2) Lagenar/cochlear recess present and elongated and tubular. The distribution of this character-state among archosauriforms is consistent across the two data matrices. An elongated and tubular cochlea recess is restricted to suchians and ornithodirans (character-state 118-1 of Nesbitt (2011), character-state 224-1 of the present analysis). The condition in ornithosuchids is unknown, and the cochlear recess is absent or short in phytosaurs and non-archosaurian archosauriforms. Therefore, this character-state also supports the hypothesis that phytosaurs lie outside Archosauria.

(3) External foramen for abducens nerves within prootic only. The scorings for this character are congruent in the two data matrices (character 122 of Nesbitt (2011), character 250 of the present analysis) and potentially support a non-crown-group position of phytosaurs. However, the optimization of this character is ambiguous at the base of Archosauria in the current analysis because in Euparkeria capensis the exit of the abducens nerves is through a foramen between the prootic and parabasisphenoid, resembling the condition in phytosaurs, whereas in erythrosuchids the exit is only within the prootic (Gower & Sennikov, 1996), resembling the condition in suchians and ornithodirans (Nesbitt, 2011).

(4) Antorbital fossa present on the lacrimal, dorsal process of the maxilla and the dorsolateral margin of the posterior process of the maxilla (the ventral border of the antorbital fenestra). The antorbital fossa occurs on the horizontal (=posterior) process of the maxilla in all ornithodirans, ornithosuchids, basal phytosaurs (e.g., Parasuchus hislopi: ISI R42; Parasuchus angustifrons: BSPG 1931 X 502), and suchians sampled here (character-state 54-2/3) (Fig. 18), but it is restricted to the ascending process of the bone in Euparkeria capensis and most proterochampsians. A similar distribution of this character-state was found by Nesbitt (2011: character 137), but he scored Parasuchus hislopi as lacking an antorbital fossa on the horizontal process of the maxilla. As a result, the distribution of this character does not unambiguously support the non-archosaurian position of phytosaurs.

(5) Posteroventral portion of the coracoid possesses a “swollen” tuber. This character-state occurs in proterochampsians, ornithodirans, ornithosuchids and suchians (character-state 401-1; Figs. 36 and 37). As a result, the absence of a swollen biceps tubercle is optimized in this analysis as an apomorphic reversal in phytosaurs. Even if phytosaurs are outside Archosauria, this character does not provide unambiguous support for this hypothesis. In this case, the optimization of the character would be ambiguous in the clade that includes proterochampsians and more crownward archosauriforms because of the absence of a well-developed biceps tubercle in Euparkeria capensis and phytosaurs and its presence in proterochampsians.

(6) Lateral tuber on the proximal portion of the ulna. The distribution of the states of this character is congruent in both analyses (character 237 of Nesbitt (2011), character 433 of the present analysis) and supports a non-archosaurian position of phytosaurs.

(7) Longest metacarpal versus longest metatarsal ratio <0.5. It is not possible to score this character confidently for any phytosaur, ornithosuchid or suchian sampled in this data matrix because of incomplete preservation (character 245 of Nesbitt (2011), character 446 of the present analysis). As a result, it could not be determined here if this character-state supports any of the competing hypotheses of phytosaur relationships.

(8) Posteromedial tuber (=anteromedial tuber of Nesbitt, 2011) of the proximal portion of the femur. The distribution of the states of this character is similar in the two data sets (character 300 of Nesbitt (2011), character 496 of the present analysis), but the posteromedial tuber of the femoral head is scored here as present in the proterochampsids Chanaresuchus bonapartei (MCZ 4035) and Gualosuchus romeri (PVL 4576) and in the basal phytosaur Parasuchus hislopi (ISI R42) (Figs. 42E and 42F). Conversely, in agreement with Nesbitt (2011), this tuber is absent in the pseudopalatine phytosaurs Nicrosaurus kapffi (SMNS 4381/1, 2) and Smilosuchus spp. (USNM 18313). As a result, the presence of a posteromedial tuber has an ambiguous optimization here and does not support unambiguously the position of phytosaurs outside archosaurs.

(9) Tibial facet of the astragalus divided into posteromedial and anterolateral basins. The scorings of this character are similar in the two analyses (character 366 of Nesbitt (2011), character 536 of the present analysis; Fig. 45). The main difference in the scorings is the absence of a subdivided tibial facet into posteromedial and anterolateral basins in Marasuchus lilloensis (PVL 3871), as scored here. In addition, Nundasuchus songeaensis possesses a gently flexed tibial facet in the astragalus (Nesbitt et al., 2014: Figs. 11G–11L) that closely resembles in morphology that of Parasuchus hislopi (ISI R42) and Smilosuchus spp. (USNM 18313). Therefore, Nundasuchus songeaensis is scored here as lacking a divided facet. As a result, the presence of a subdivided tibial facet in the astragalus is optimized here as an autapomorphy of Lagerpeton chanarensis and a synapomorphy of the clade composed of ornithosuchids and suchians. It does not therefore support a non-archosaurian position of phytosaurs.

(10) Calcaneal tuber orientation, relative to the transverse plane, between 50°and 90°posteriorly. The distribution of the states of this character is congruent in the two analyses (character 377 of Nesbitt (2011), character 546 of the present analysis; Fig. 45) and is consistent with a non-archosaurian position of phytosaurs but also of Nundasuchus songeaensis, which has been recently interepreted as a basal pseudosuchian archosaur by Nesbitt et al. (2014).

In summary, there is a lot of congruence between the relevant scorings to reconstruct the phylogenetic relationships of phytosaurs in this and Nesbitt’s (2011) data matrices, and several characters support the hypothesis of phytosaurs outside Archosauria. However, several other apomorphic features in this analysis support the inclusion of phytosaurs within Archosauria (e.g., posttemporal fenestra smaller than the supraoccipital but does not develop as a small foramen; posterior process of the squamosal ventrally curved; anterior ramus of the pterygoid transversely narrow along its entire extension; posteroventral process of the dentary contributes to the border of the external mandibular fenestra; articular with a ventromedially directed process; spine table at the distal end of the postaxial cervical and dorsal neural spines; fibular condyle projecting distally distinctly beyond the tibial condyle in the distal end of the femur; area of attachment of the iliofibularis muscle on the fibula on a hypertrophied tubercle; calcaneal tuber approximately as broad as or broader than tall at midshaft; calcaneal tuber with a expanded distal end in proximal or distal view; pedal unguals strongly transversely compressed, with a sharp dorsal keel). Indeed, the analysis of Nesbitt et al. (2014) found a Bremer support of 1 for the clade including avemetatarsalians and pseudosuchians but not phytosaurs. In conclusion, the most parsimonious hypothesis found here is the monophyly of a traditional Archosauria including phytosaurs as pseudosuchians, but more research is needed on the reconstruction of the higher-level phylogenetic relationships of phytosaurs because of the presence of conflicting phylogenetic evidence—for current hypotheses—in this part of the archosauromorph tree. Because of the long ghost lineage at the base of Phytosauria (Nesbitt, 2011), the discovery of new Early and Middle Triassic basal archosaurs and their most immediate sister-taxa and especially pre-Late Triassic phytosaurs is crucial in order to shed light on this issue.

The phylogenetic position of Nundasuchus

Nundasuchus songeaensis is a recently described archosauriform from the Middle Triassic of Tanzania (Nesbitt et al., 2014). Nesbitt et al. (2014) stated that Nundasuchus songeaensis possesses a combination of character-states typical of some of the earliest diverging archosaurs as predicted by both Nesbitt (2011) and Brusatte et al. (2010). Nesbitt et al. (2014) recovered Nundasuchus songeaensis as one of the most basal suchians using a modified version of the data set of Nesbitt (2011), being the sister-taxon of the clade composed of Ticinosuchus ferox and paracrocodylomorphs. By contrast, when these authors used a modified version of the data matrix of Brusatte et al. (2010), Nundasuchus songeaensis was found as the sister-taxon of all other pseudosuchians. Nesbitt et al. (2014) highlighted that the combination of archosaur plesiomorphies and syanpomorphies found only within Pseudosuchia (excluding phytosaurs) make difficult to decisively determine whether Nundasuchus songeaensis represents an early suchian or a more basal pseudosuchian, or if the taxon belongs within Archosauria at all.

Nundasuchus songeaensis is found is this analysis as the sister-taxon of the clade composed of ornithosuchids and suchians, thus representing a position incongruent with the result recovered by Nesbitt et al. (2014) when they used the modified version of the data matrix of Nesbitt (2011). By contrast, the position found here for the species is congruent with that found in the modified version of the data matrix of Brusatte et al. (2010) (Nesbitt et al., 2014). The clade formed by Nundasuchus songeaensis, ornithosuchids, and suchians is supported by four postcranial synapomorphies (see synapomorphies of the Nundasuchus songeaensis + Suchia clade in Results) and possesses a Bremer support of 2 (without the pruning of taxa with a large amount of missing data) and absolute and GC bootstrap frequencies of 16% and 3%, respectively. Five postcranial synapomorphies exclude Nundasuchus songeaensis from the clade composed of ornithosuchids and suchians and this clade has a Bremer support of 2 and absolute and GC bootstrap frequencies of 13% and 0%, respectively (without the pruning of taxa with a large amount of missing data). Two additional steps are necessary to force the position of Nundasuchus songeaensis as the sister-taxon of the paracrocodylomorphs sampled here or as the sister-taxon of all other pseudosuchians.

Contrasting with the results recovered here, Nesbitt et al. (2014) found three synapomorphies that support the clade Nundasuchus songeaensis + Ticinosuchus ferox + Paracrocodylomorpha that are discussed as follows:

(1) Distal end of neural spines of the cervical vertebrae expanded anteriorly so that the spine table is triangular or heart-shaped in dorsal view. This character-state is represented in the character 323 of this analysis and character 191 of Nesbitt (2011). Both data sets present very consistent scorings for this character among the sampled phytosaurs, ornithosuchids and suchians. However, the phytosaurs Parasuchus hislopi (ISI R42) and Smilosuchus spp. (USNM 18313; Camp, 1930) and the paracrocodylomorph Batrachotomus kupferzellensis (Gower & Schoch, 2009; SMNS specimens) has been scored as polymorphic here because of the presence of both sub-oval/sub-rectangular and sub-triangular/heart-shaped neural spines in dorsal view along the cervical and dorsal vertebral series. Nundasuchus songeaensis is scored as possessing sub-triangular/heart-shaped cervical and dorsal neural spines in dorsal view, whereas the paracrocodylomorph Prestosuchus chiniquensis (UFRGS-PV-0152-T, 0156-T, 0629-T) and other pseudosuchians are scored as lacking this condition (Riojasuchus tenuisceps: PVL 3827; Turfanosuchus dabanensis: IVPP V3237; Gracilisuchus stipanicorum: MCZ 4118, PULR 08; Aetosauroides scagliai: PVL 2073). As a result, the presence of triangular or heart-shaped presacral neural spines in dorsal view is optimized here as an autapomorphy of Nundasuchus songeaensis rather than a synapomorphy of a less inclusive clade within Suchia. However, this result may be an artefact because of the poor sampling of suchians in this data matrix. For example, the following suchians that present the same character-state as Nundasuchus songeaensis are missing from this data set: Revueltosaurus callenderi, Saurosuchus galilei and Postosuchus alisonae.

(2) Straight transverse groove present on the proximal surface of the femur. The scorings of this character are consistent in both data matrixes among archosaurs (character 314 of Nesbitt (2011) and character 495 of this data set; Fig. 42). Within the sample pseudosuchians of this analysis, only Nundasuchus songeaensis, Ornithosuchus longidens and the paracrocodylomorphs Prestosuchus chiniquensis and Batrachotomus kupferzellensis possess a straight transverse groove on the proximal surface of the femur. As a result, the phylogenetic signal of this character is congruent with the hypothesis that Nundasuchus songeaensis is more closely related to paracrocodylomorphs than to other archosaurs.

(3) Dorsal osteoderms with a staggered alignment dorsal to the presacral vertebrae. Within Archosauria, only Nundasuchus songeaensis and Gracilisuchus stipanicicorum (MCZ 4118, PULR 08) are scored here to have a staggered alignment of the presacral dorsal osteoderms (character-state 411-1 of Nesbitt (2011) and character-state 594-0 of this data matrix). Nesbitt (2011) scored the presence of this character-state also in the referred specimen of Prestosuchus chiniquensis UFRGS-PV-0156-T, but after a re-examination of this specimen it is reinterpreted that the alignment of the dorsal osteoderms is one to one. The latter condition is also present in Parasuchus hislopi (ISI R42), ornithosuchids (Ornithosuchus longidens: Walker, 1964; Riojasuchus tenuisceps: PVL 3827), Aetosauroides scagliai (PVL 2059, 2073) and Batrachotomus kupferzellensis (Gower & Schoch, 2009), in agreement with the scorings of Nesbitt (2011). As a result, the distribution of the scoring of this character under the current taxonomic sample does not support the hypothesis that Nundasuchus songeaensis is more closely related to paracrocodylomorphs than to other archosaurs. However, an improved taxonomic sample of suchians may favour the latter hypothesis.

Therefore, the phylogenetic position recovered here for Nundasuchus songeaensis as the sister-taxon of all pseudosuchians with the exception of phytosaurs is congruent with the suite of plesiomorphic character-states present in this species (e.g., proximal tarsals; Nesbitt et al., 2014). Nevertheless, the phylogenetic position of Nundasuchus songeaensis is not strongly supported in this analysis and the increase of the taxonomic sample of suchians might easily change the relationships of this species. The phylogenetic position of this animal needs further testing in future analyses.

The phylogenetic position of Pterosauria

The vast majority of quantitative phylogenetic analyses found pterosaurs as members of the bird-line of the crown-group Archosauria (e.g., Gauthier, 1986; Benton, 1990) and in most cases as the sister-taxon of Dinosauromorpha (e.g., Sereno, 1991; Brusatte et al., 2010; Ezcurra, Lecuona & Martinelli, 2010; Nesbitt, 2011). However, the pioneering quantitative phylogenetic analysis of Benton (1985) recovered pterosaurs as the sister-taxon of all other archosauromorphs, but this hypothesis has been subsequently rejected (e.g., Benton, 1990; Sereno, 1991; Nesbitt, 2011). In more recent times, Bennett (1996) found pterosaurs as the sister-taxa of the clade composed of Eythrosuchidae and Eucrocopoda. A congruent topology has been recently recovered again by the same author, in which pterosaurs are nested in a trichotomy together with erythrosuchids and eucrocopods (Bennett, 2012). An alternative phylogenetic position for pterosaurs has been proposed by Peters (2000), in which pterosaurs are non-archosauriform archosauromorphs nested within Tanystropheidae (sensu the definition of the clade by Dilkes (1998) that includes Macrocnemus as an internal specifer, contra Peters (2000)). The phylogenetic analysis conducted here constitutes the best data matrix compiled so far to test the position of pterosaurs within Archosauromorpha because of the broad sample of Permo-Triassic species, including the undoubted pterosaur Dimorphodon macronyx.

The result of this analysis strongly supports the position of pterosaurs (represented here only by Dimorphodon macronyx) in the bird-line of archosaurs, in agreement with the vast majority of analyses conducted in the last three decades. Indeed, the monophyly of Ornithodira is supported by 19 synapomorphies and the branch possesses a Bremer support of 13 (following the pruning of terminals with a large amount of missing data) and absolute and GC bootstrap frequencies of 87% and 85%, respectively. Thirty-six additional steps are necessary to force the position of Dimorphodon macronyx as the sister-taxon of Tanystropheidae or as a member of this clade, and 33 extra steps to place Dimorphodon macronyx as the sister-taxon of erythrosuchids and eucrocopods under topologically constrained searches. The optimal alternative position that Dimorphodon macronyx can adopt in this analysis outside the bird-line of Archosauria is as the most basal pseudosuchian or phytosaur, and this topology is 14 steps less optimal than the original MPTs. Therefore, the hypothesis of a non-monophyletic Ornithodira—considering Dimorphodon macronyx as the only floating taxon—is rejected here by a substantial amount of evidence. Future analyses focused on testing the higher-level phylogenetic relationships of pterosaurs should also incorporate a broader sample of early pterosaurs and some enigmatic diapsids that were found as more closely related to pterosaurs than to other archosauromorphs by Peters (2000) and are not included in the current taxonomic sample (i.e., Langobardisaurus pandolfi, Cosesaurus aviceps, Sharovipteryx mirabilis and Longisquama insignis). However, it seems extremely unlikely that the addition of these enigmatic diapsids, which are unambiguously considered to not be members of Archosauriformes (e.g., Peters, 2000; Senter, 2004), will affect the higher-level phylogenetic position of pterosaurs.

Homoplasy in character-state transformations

Among the twelve anatomical regions, the higher means of the consistency index (CI) values (CI > 0.50) are found in the characters of the braincase and hindlimb (Table 3), although they do not differ significantly from other anatomical regions (with the exception of a significant difference between the lower jaw and hindlimb means). These higher means coincide with anatomical areas (e.g., braincase, tarsus) that have been considered by previous workers as among the most phylogenetically informative regions (e.g., Gauthier, Kluge & Rowe, 1988; Sereno & Arcucci, 1990; Parrish, 1993; Gower & Sennikov, 1996). Therefore, this result bolsters partially these previous observations. Conversely, the lowest means are observed in lower jaw, marginal dentition, and sacral and caudal vertebral characters (CI < 0.40). The marginal dentition is the only region that does not possess a distinct bimodal distribution between extreme values, with the CI values ranging from 0.1 to 0.5. This result is also in agreement with previous claims of high homoplasy rates and rampant convergence in dental characters within archosauromorph evolution (e.g., Irmis et al., 2007b; Hwang, 2011). The lower CI values present in marginal dentition characters may be related to higher selective pressures and/or evolutionary rates because of being directly related to trophic habits. The presence of more homoplastic characters in the lower jaw, marginal dentition, and sacral and caudal vertebrae does not mean that they are directly less phylogenetically informative than other regions, but they may be informative at lower taxonomic levels, as is demonstrated by the recovery of several synapomorphies that belong to these anatomical regions in the early archosauromorph tree. However, phylogenetic assignments based mainly on character-states from these anatomical regions should be taken with caution, as well as the systematic interpretation of specimens limited to elements of these regions, such as isolated lower jaw bones or teeth (e.g., Parker et al., 2005; Irmis et al., 2007b; Heckert & Miller-Camp, 2013).

Table 3 Number of characters, mean and standard deviation of the consistency indexes of the twelve anatomical groups.

Group	N	Mean (standard deviation)	
Dermal skull bones	173	0.42 (±0.28)	
Palatoquadrate	32	0.47 (±0.32)	
Braincase	54	0.52 (±0.31)	
Lower jaw	36	0.36 (±0.30)	
Marginal dentition	11	0.34 (±0.15)	
Presacral vertebrae and ribs	58	0.40 (±0.29)	
Sacral and caudal vertebrae and ribs	14	0.32 (±0.20)	
Scapular girdle	30	0.41 (±0.33)	
Forelimb	40	0.45 (±0.27)	
Pelvic girdle	34	0.40 (±0.22)	
Hindlimb	97	0.53 (±0.30)	
Dermal scutes	13	0.40 (±0.27)	

Macroevolutionary implications

The phylogenetic analyses presented here include the most extensive sampling of Permo-Triassic non-archosaurian archosauromorphs published to date, and the results have important implications for understanding the taxonomy, biogeography, and timming and mode of early archosauromorph evolution. The origin and early evolution of archosauromorphs during the Permian have been already discussed by Ezcurra, Scheyer & Butler (2014) and Bernardi et al. (2015), and the most relevant macroevolutionary implications of the results found in this phylogenetic analysis are discussed here.

Figure 61 Time-calibrated phylogenetic tree of non-eucrocopodan archosauromorphs recovered in this analysis.

Eucrocopodans have been merged into a single terminal in this tree. The length of the vertical bar representing each terminal taxon represents chronostratigraphical uncertainty rather than true stratigraphical range. Abbreviations: Ans, Anisian; Cap, Capitanian; Chx, Changhsingian; Crn, Carnian; E., Early; Ind, Induan; Lad, Ladinian; Nor, Norian; Ole, Olenekian; Rht, Rhaetian; Roa, Roadian; Wor, Wordian; Wuc, Wuchiapingian. Geological timescale after Gradstein et al. (2012).

Figure 62 Time-calibrated phylogenetic tree of eucrocopodan archosauromorphs recovered in this analysis.

Non-eucrocopodan archosauromorphs are not shown in this tree. The length of the vertical bar representing each terminal taxon represents chronostratigraphical uncertainty rather than true stratigraphical range. Abbreviations: Ans, Anisian; Cap, Capitanian; Chx, Changhsingian; Crn, Carnian; E., Early; Ind, Induan; Lad, Ladinian; Nor, Norian; Ole, Olenekian; Rht, Rhaetian; Roa, Roadian; Wor, Wordian; Wuc, Wuchiapingian. Geological timescale after Gradstein et al. (2012).

The diversity of non-archosauriform archosauromorph lineages. The higher-level taxonomic diversity of non-archosauriform archosauromorphs has been usually considered to be mainly restricted to two diverse groups, namely “Prolacertiformes” and Rhynchosauria. However, a broadly polyphyletic “Prolacertiformes” sensu Jalil (1997) is recovered in the phylogenetic analysis conducted here, in which putative prolacertiform species are distributed among five different lineages of non-archosauriform archosauromorphs. Therefore, the temporal calibration of the topology recovered here implies that at least eight independent archosauromorph clades crossed the Permo-Triassic boundary (Figs. 61 and 62). These independent lineages of non-archosauriform archosauromorphs were morphologically and probably ecologically disparate during the Triassic, including gracile, long-necked aquatic forms (tanystropheids), bulky and long-necked terrestrial herbivores (allokotosaurians), gracile and long-necked predatory species (e.g., prolacertids), and hyperspecialized herbivores (rhynchosaurs). This palaeoecological diversity among non-archosauriform archosauromorphs seems to exceed that present in coeval non-archosaurian archosauriforms, which are mainly represented by crocodile-like (e.g., proterosuchids, proterochampsids, doswelliids) and massive (e.g., erythrosuchids) predatory clades. These observations should be tested in the future by quantitative macroevolutionary analyses.

Proterosuchidae as a short-lived “disaster” clade, and the biotic recovery after the Permo-Triassic mass extinction. The biochron of the proterosuchids has been previously suggested to to range from the latest Permian (Archosaurus rossicus) to the late Anisian (“Chasmatosaurus ultimus”) (Charig & Reig, 1970; Charig & Sues, 1976; Gower & Sennikov, 2000; Ezcurra, Butler & Gower, 2013). However, based on the present phylogenetic analysis proterosuchids are much more restricted temporally and taxonomically than previously thought. Proterosuchids as conceived here are known from the latest Permian of Russia and the earliest Triassic (Induan) of South Africa and China and ambiguously from Russia and India (Figs. 4 and 61). As a result, current evidence indicates that proterosuchids are a short-lived clade that is documented in the fossil record for a period of probably less than three million years. In particular, proterosuchids are restricted to a short stratigraphic section of 5–14 metres above the Permo-Triassic boundary in the Lystrosaurus AZ of South Africa, and they disappear during the first recovery phase of the extinction event (Smith & Botha-Brink, 2014). The proterosuchid-bearing levels of China are not as well stratigraphically constrained as those of South Africa, and it is possible that the biostratigraphic ranges of the proterosuchids in these horizons are also limited to the first few metres above the Permo-Triassic boundary. Therefore, proterosuchids potentially represent a component of the initial recovery phase after the extinction event, which in the case of the archosauriforms seems to be characterized by a high taxonomic diversity of morphologically rather similar species (Ezcurra & Butler, 2015a). The biostratigraphic range of proterosuchids closely resembles, but it is even more chronostratigraphically restricted than, that of the dicynodont genus Lystrosaurus (Smith & Botha-Brink, 2014).

Morphologically disparate archosauriform groups are documented in Olenekian beds for the first time in the fossil record, including “intermediate” forms between proterosuchids and erythrosuchids (e.g., Chasmatosuchus rossicus, Fugusuchus hejiapaensis), erythrosuchids (e.g., Garjainia madiba, Garjainia prima) and ctenosauriscid archosaurs (e.g., Vytshegdosuchus zbeshartensis, Ctenosauriscus koeneni) (Gower & Sennikov, 2000; Butler et al., 2011). However, the occurrence of ctenosauriscids that are deeply nested within Archosauria indicates that, at least, all other major known non-archosaurian archosauriform clades should have been present by that time (Butler et al., 2011). There is no current evidence that the initial diversification of archosauriforms occurred immediately after the mass extinction event (i.e., during the Lystrosaurus AZ) and this diversification was perhaps delayed between 1 and 5 million years after the mass extinction. As a result, the evolutionary history of archosauriforms during the Early Triassic can be subdivided in a first phase characterized by the short-lived “disaster-clade” Proterosuchidae and a second phase that witnessed the initial morphological and probably palaeoecological diversification of the group.

Comments on the evolutionary history of Erythrosuchidae. The oldest erythrosuchids come from the late Olenekian of South Africa and Russia (Garjainia prima and Garjainia madiba; Ochev, 1958; Gower et al., 2014) and possibly from China and Australia (Peng, 1991; this study). Therefore, there is no current evidence for a temporal overlap between unambiguous proterosuchids and erythrosuchids, contrasting with previous hypotheses (Charig & Reig, 1970; Charig & Sues, 1976; Ezcurra, Butler & Gower, 2013). The taxonomic content and temporal distribution of erythrosuchids recovered here is similar to those of previous interpretations (Kalandadze & Sennikov, 1985; Gower & Sennikov, 2000), and the group ranges from the late Olenekian to the Ladinian, thus accounting for an evolutionary history of approximately 12 million years. Therefore, the biochron of erythrosuchids considerably exceeds that of proterosuchids.

The general skull shape of Guchengosuchus shiguaiensis is fairly unknown because of the fragmentary condition of the holotype and only known specimen and, unfortunately, obscures understanding of the ancestral cranial morphology of the clade. However, all the remaining and more deeply nested erythrosuchids possess a consistent dorsoventrally tall, massive skull with large marginal teeth (e.g., Garjainia prima, Erythrosuchus africanus, Shansisuchus shansisuchus). The cranial anatomy of erythrosuchids closely resembles that of rauisuchian archosaurs (e.g., Batrachotomus kupferzellensis, Saurosuchus galilei) and these similarities have been interpreted as convergences resulting from similar predatory habits. Indeed, the convergences between erythrosuchids and rauisuchians nourished systematic debates, such as the disputed phylogenetic position of Youngosuchus sinensis as an erythrosuchid or a suchian archosaur (Young, 1973b; Kalandadze & Sennikov, 1985; Parrish, 1992). Beyond the overall similarities between the erythrosuchid and rauisuchian skull morphology, some erythrosuchids possess an unusual apomorphy unrecorded in other Permo-Triassic archosauromorphs, the presence of a secondary antorbital fenestra between the postnarial process of the premaxilla and the anterior margin of the maxilla. This accessory opening is present in Guchengosuchus shiguaiensis (Peng, 1991), Shansisuchus shansisuchus (Young, 1964; Wang et al., 2013) and Chalishevia cothurnata (Ochev, 1980). However, it is very interesting to note that the internal phylogenetic relationships for the erythrosuchids recovered here imply that the secondary antorbital fenestra was independently acquired by Guchengosuchus shiguaiensis and the clade composed of Shansisuchus shansisuchus and Chalishevia cothurnata.

The appendicular anatomy of erythrosuchids closely resembles that present in allokotosaurians, rhynchosaurs, prolacertids, and proterosuchids, with poorly ossified epiphyses on the long bones and a distinctly sprawling gait (Gower, 2003; Ezcurra, Butler & Gower, 2013). Therefore, the evolutionary history of Erythrosuchidae seems to have been characterized by the acquisition of several apomorphies mostly restricted to the skull, but the retention of a relatively unspecialized appendicular skeleton. Thus, despite the similarities between the skulls of erythrosuchids and rauisuchians, which probably indicate that both groups played the role of apex predators in their ecosystems (Sennikov, 1996), their palaeocology was probably rather different (e.g., locomotion) and this hypothesis should be tested in the future by biomechanical analyses.

Proterochampsia: a diverse clade of non-archosaurian archosauriforms with aquatic adaptations. The present phylogenetic analysis recovered a taxonomically rich monophyletic group of non-crown-group archosauriforms with aquatic adaptations, which is composed of at least 15 nominal species (13 of them sampled in the current analysis) (Trotteyn, Arcucci & Raugust, 2013; Sues, Desojo & Ezcurra, 2013). Several of the members of this clade possess characters that have been previously interpreted as evidence of a semi-aquatic (e.g., dorsally facing external nares and orbits; Romer, 1971a; Trotteyn, Arcucci & Raugust, 2013; Sues, Desojo & Ezcurra, 2013) to a fully aquatic mode of life (e.g., reduced limbs; Nesbitt et al., 2009) (but the mode of life of the South American doswelliids is unknown). The evolutionary radiation of Proterochampsia indicates that the invasion of continental aquatic niches by non-archosaurian or non-phytosaurian archosauriforms was more successful than previously thought and was probably restricted to a single major lineage. Previous hypotheses of semi aquatic habits for proterosuchids and erythrosuchids have been recently considered unlikely based on palaeohistological evidence (Botha-Brink & Smith, 2011). The morphological and taxonomic diversification of proterochampsians is probably part of a third phase in archosauriform evolution during the Triassic, which also included the invasion of the marine realm by some poposauroids (Nesbitt et al., 2013) and the appearance of herbivorous clades, such as aetosaurs (Desojo et al., 2013), during the late Middle to Late Triassic (Fig. 62). These simplified three phases described here during the early evolution of archosauriforms lead to the numerical dominance of the group by the latest Triassic, which seems to have been enhanced during the Early Jurassic by the empty niches left by the Triassic-Jurassic mass extinction (Brusatte et al., 2008).

Prospectus

The phylogenetic relationships of multiple lineages of non-archosaurian archosauromorphs, with a special emphasis on proterosuchian archosauriforms, were explored through the most comprehensive data matrix published to date. This study includes for the first time several archosauromorph species in a quantitative analysis and, as a result, it allows rigorous testing of the monophyly of some key taxa crucial to understanding the early evolutionary history of the group that gave rise to non-avian dinosaurs, birds, and crocodylians. The data matrix compiled here and the results obtained from it should serve as the basis for future research on the phylogenetic relationships of Permo-Triassic archosauromorphs. This analysis should be also used as the starting point to build a comprehensive phylogeny of Permo-Triassic archosauromorphs, which will allow exploring and testing quantitatively macroevolutionary and palaeobiogeographical hypotheses within a broad taxonomic sample and well estabslihed phylogenetic framework.

The results of this study strongly bolster the monophyly of several archosauromorph clades, such as Allokotosauria, Rhynchosauria, Erythrosuchidae, Proterochampsia, Avemetatarsalia, Phytosauria, and Riojasuchidae (with Bremer supports of ≥10). Similarly, the higher-level phylogenetic relationships of non-archosaurian archosauromorphs also show some very well supported clades, including Archosauriformes, the clade composed of Eythrosuchidae + more crownward archosauriforms, and Eucrocopoda (with Bremer supports of ≥12). However, the support for some higher-level clades is rather weak, indicating where future research efforts should be focused, such as in the interrelationships between non-archosauriform crocopods, the position of phytosaurs as non-archosaurian eucrocopods or basal pseudosuchians, and the position of Nundasuchus songeaensis within Pseudosuchia. This phylogenetic analysis comprehensively sampled non-archosaurian archosauriform groups, such as proterosuchids, erythrosuchids, and proterochampsids. Nevertheless, future analyses will need to improve the taxonomic sample of Triassic species of clades such as tanystropheids, allokotosaurians, rhynchosaurs, and archosaurs in order to explore the first 60 million years of macroevolutionary history of Archosauromorpha as a whole. This kind of comprehensive analyses will be paramount to understand the timing and mode of evolution of archosauromorphs during a key moment in the history of life, which witnessesed two of the most severe biotic crises documented in the fossil record and the origin of the main modern vertebrate groups.

Supplemental Information

Appendix S1 Calculated ratios for the discretization of meristic characters.

Click here for additional data file.

Appendix S2 Data matrix in Nexus file.

Click here for additional data file.

Appendix S3 Data matrix in TNT file.

Click here for additional data file.

This paper is part of my PhD thesis that was conducted initially at Ludwig-Maximilians-Universität München (Germany) and completed at the University of Birmingham (United Kingdom). I thank my supervisor Richard Butler for suggesting this line of research, continuous advice and encouragement, and providing funding from grants coordinated by him during the last three years. I am also very thankful for all the comments and critical suggestions of my external co-supervisor David Gower. I thank Roland Sookias for discussions about basal archosauromorph anatomy and systematics and help during our collection visits in the last three years. I appreciate the comments of my internal co-supervisor Ivan Sansom, which were very useful during the second half of my PhD. I thank Isabella von Lichtan for preparing casts of some of the bones of Tasmaniosaurus triassicus. I thank Fernando Abdala for unpublished pictures of RC 846 and Bhart-Anjan Bhullar for allowing me access to unpublished CT data of the same specimen. I thank Sterling Nesbitt and Mariana Grassetti for some photographs used in Fig. 57. I thank technical staff of the QM for additional preparation conducted on specimens of Kalisuchus rewanensis. The loan of the Prolacerta broomi specimen GHG 431 was possible thanks to Ellen de Kock (GHG). I thank Andrey Sennikov for discussion and useful information about the locality of Archosaurus rossicus. Discussions with Fernando Abdala, Federico Agnolín, Belén von Baczko, Michael Benton, Jennifer Botha-Brink, Juan Carlos Cisneros, Julia Desojo, Christian Foth, Jason Hilton, Christian Kammerer, Adriana López-Arbarello, Sterling Nesbitt, Grzegorz Niedźwiedzki, Oliver Rauhut, Bruce Rubidge, Jörg Schneider, Andrey Sennikov and Michael Szurlies are appreciated and were very useful at different stages of this research. The suggestions and comments of the reviewers Nick Fraser and, specially, Adam Pritchard are strongly appreciated and have been found very useful to improve the quality of the final version of the manuscript. I am very grateful to Hans-Dieter Sues for his editorial work, which improved the final version of the manuscript. Access to the free version of TNT 1.1 was possible due to the Willi Henning Society.

I thank the following curators, researchers and collection managers that provided access to specimens under their care for the purpose of this research: Billy de Klerk (AM); Carl Mehling and Steve Brusatte (AMNH); Matt Williams (BRLSI); Bernhard Zipfel, Bruce Rubidge, Jonah Choiniere and Fernando Abdala (BP); Markus Moser and Oliver Rauhut (BSPG); Lucas Fiorelli (CRILAR); H. Morgenroth and R. Taylor (EXEMS); William Simpson (FMNH); Ellen de Kock (GHG); curatorial staff of the GSI (Kolkata); Saswati Bandyopadhyay and Dhurjati Sengupta (ISI); Jun Liu and Corwin Sullivan (IVPP); Alejandro Kramarz and Stella Alvarez (MACN-Pv); Daniela Schwarz (MB); Susana Devinvenzi González (MCNAM); Claudia Malabarba (MCP); Jessica Cundiff (MCZ); Ronan Allain (MNHN); Sandra Chapman and Lorna Steel (NHMUK PV); Elize Butler and Jennifer Botha-Brink (NMQR); Heinz Furrer (PIMUZ); Andrey Sennikov (PIN); Emilio Vaccari and Gabriela Cisterna (PULR); Jaime Powell (PVL); Ricardo Martínez (PVSJ); Kristen Spring and technical staff (QM); Mary-Anne Binnie (SAM); Sheena Kaal and Roger Smith (SAM-PK); Daniel Lockett (SHYMS); Rainer Schoch (SMNS); Heidi Fourie (TM); Kevin Padian, Pat Holroyd and Randall Irmis (UCMP); César L. Schultz (UFRGS); Atila S. Da-Rosa (UFSM); Mathew Lowe and Jennifer Clack (UMZC); Michael Brett-Surman and Hans-Dieter Sues (USNM); Isabella von Lichtan (UTGD); Jon Radley (WARMS); and Mateusz Talanda and Tomasz Sulej (ZPAL).

Institutional Abbreviations

AM Albany Museum, Grahamstown, South Africa

AMNH American Museum of Natural History, New York, USA

AUP University of Aberdeen, Palaeontology collection, Aberdeen, UK

BRLSI (formerly BATGM) Bath Royal Literary and Scientific Institution, Bath, UK

BP Evolutionary Studies Institute (formerly Bernard Price Institute for Palaeontological Research), University of the Witwatersrand, Johannesburg, South Africa

BRSUG University of Bristol, School of Earth Sciences, Bristol, United Kingdom

BSPG Bayerische Staatssammlung für Paläontologie und Geologie, Munich, Germany

CA Colégio Anchieta, Porto Alegre, Brazil

CPEZ Coleção Municipal, São Pedro do Sul, Brazil

CRILAR Centro Regional de Investigaciones y Transferencia Tecnológica de La Rioja, Paleontología de Vertebrados, Anillaco, La Rioja, Argentina

EM Elgin Museum, Elgin, UK

EXEMS Royal Albert Memorial Museum, Exeter, UK

FG Paläontologische und Stratigraphische Sammlung an der Bergakademie Freiberg im Humboldt-Bau, Freiburg, Germany

FMNH Field Museum of Natural History, Chicago, USA

GHG Geological Survey, Pretoria, South Africa

GMB Geological Institute, Beijing, China

GPIT Paläontologische Sammlung der Universität Tübingen, Tübingen, Germany

GR Ghost Ranch Ruth May Museum of Paleontology, New Mexico, USA

GSI Geological Survey of India, Kolkata, India

GSM Geological Survey and Museum, Kcyworth, UK

IGCAGS Institute of Geology, Chinese Academy of Geological Sciences, Beijing, China

IPB Institut für Paläontologie, Universität Bonn, Bonn, Germany

ISI Indian Statistical Institute, Kolkata, India

IVPP Institute of Vertebrate Paleontology and Paleoanthropology, Beijing, China

KUVP Kansas University Museum of Natural History, Lawrence, USA

MACN-He Museo Argentino de Ciencias Naturales “Bernardino Rivadavia”, Herpetología, Buenos Aires, Argentina

MACN-Pv Museo Argentino de Ciencias Naturales “Bernardino Rivadavia”, Paleontología de Vertebrados, Buenos Aires, Argentina

MANCH Manchester Museum, Manchester, UK

MCNAM Museo de Ciencias Naturales y Antropológicas de Mendoza (J. C. Moyano), Mendoza, Argentina

MCP Museu de Ciências e Tecnologia da Pontifícia Universidade Católica do Rio Grande do Sul, Porto Alegre, Brazil

MCZ Museum of Comparative Zoology, Cambridge, USA

MHI Muschelkalkmuseum, Ingelfingen, Germany

MLP Museo de La Plata, La Plata, Argentina

MNA Museum of Northern Arizona, Flagstaff, USA

MNHN Muséum national d’Histoire naturelle, Paris, France

MSNM Museo di Storia Naturale, Milano, Italy

Nat. Kab. Naturalienkabinett und Heimatmuseum, Waldenburg, Germany

NHMUK PV The Natural History Museum, Palaeontology Vertebrates, London, UK

NHMW Naturhistorisches Museum Wien, Vienna, Austria

NMK Naturkundemuseum im Ottoneum, Kassel, Germany

NMQR National Museum, Bloemfontein, South Africa

NMT National Museum of Tanzania, Dar es Salaam, Tanzania

PEFO Petrified Forest National Park, Arizona, USA

PIMUZ Paläontologisches Institut und Museum der Universität Zürich, Zurich, Switzerland

PIN Paleontological Institute of the Russian Academy of Sciences, Moscow, Russia

PPHM Panhandle-Plains Historical Museum, Canyon, USA

PSM Privatsammlung W. Munk, Walzbachtal, Germany

PULR Paleontología, Universidad Nacional de La Rioja, La Rioja, Argentina

PVL Paleontología de Vertebrados, Instituto ‘Miguel Lillo’, San Miguel de Tucumán, Argentina

PVSJ División de Paleontología de Vertebrados del Museo de Ciencias Naturales y Universidad Nacional de San Juan, San Juan, Argentina

QM Queensland Museum, Brisbane, Queensland, Australia

RC Rubidge Collection, Wellwood, Graaff-Reinet, South Africa

SAM South Australian Museum, Adelaide, Australia

SAM-PK Iziko South African Museum, Cape Town, South Africa

SHYMS Shropshire Museum, Lodlow, UK

SMNS Staatliches Museum für Naturkunde Stuttgart, Stuttgart, Germany

SXMG Shanxi Museum of Geology, Taiyuan, Shanxi, China

TM Ditsong National Museum of Natural History (formerly Transvaal Museum), Pretoria, South Africa

TMM Texas Memorial Museum, Austin, USA

UA University of Antananarivo, Antananarivo, Madagascar

UCL University College London, London, UK

UCMP University of California Museum of Paleontology, Berkeley, USA

UFRGS Universidade Federal do Rio Grande do Sul, Porto Alegre, RS, Brazil

UFSM Universidade Federal do Rio Grande do Sul, Porto Alegre, Brazil

UMMP University of Michigan Museum of Paleontology, Ann Arbor, USA

UMZC University Museum of Zoology, Cambridge, UK

USNM National Museum of Natural History (formerly United States National Museum), Smithsonian Institution, Washington, D.C., USA

UTGD School of Earth Sciences, University of Tasmania, Hobart, Australia

WARMS Warwickshire Museum, Warwick, UK

WMsN Westfälisches Museum für Naturkunde, Münster, Germany

YPM Yale Peabody Museum, New Haven, Connecticut, USA

ZAR Muséum national d’Histoire naturelle (Zarzaitine collection), Paris, France

ZMR MB Museum für Naturkunde–Leibniz-Institut für Evolutions- und Biodiversitätsforschung, Berlin, Germany

ZPAL Institute of Paleobiology of the Polish Academy of Sciences, Warsaw, Poland

Additional Information and Declarations

Competing Interests

Author Contributions

Data Availability

The author declare there are no competing interests.

Martín D. Ezcurra conceived and designed the experiments, performed the experiments, analyzed the data, contributed reagents/materials/analysis tools, wrote the paper, prepared figures and/or tables, reviewed drafts of the paper.

The following information was supplied regarding data availability:

Morphobank Project 2416: http://morphobank.org/index.php/Projects/ProjectOverview/project_id/2416

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
