# Peer review of "The phylogenetic relationships of basal archosauromorphs, with an emphasis on the systematics of proterosuchian archosauriforms"

_PeerJ, doi:10.7717/peerj.1778_

## Round 0.1 · original submission · Minor Revisions

· Academic Editor

Minor Revisions

Please address the comments by both referees in detail. It is also important to revew the entire text again for spelling and grammar mistakes because the journal does not provide copy-editing.

·

Basic reporting

I only have a number of small comments on phrasing and spelling. I've also noted several characters that require a few more exemplar taxa in the descriptions.

Experimental design

I have only minor comments on the phylogenetic characters developed by the author. I have also made some small comments on codings throughout the text. I would ask that the author comment on these differences rather than rerun all of the analyses for his revisions.

Validity of the findings

No comments.

Additional comments

This phylogenetic study represents a sorely needed expansion of our understanding of the clade Archosauromorpha, focused on primarily on “proterosuchians.” It’s excellent to see an extensive revision of early archosauriforms, especially when so many geographically and stratigraphically important taxa are not represented in phylogenetic studies. The extensive descriptions and illustrations of phylogenetic characters will be quite useful for future work. Of particular value are the high-quality photographs and updated descriptions of a number of early archosauromorphs that have never been studied in a modern context. Overall, this paper represents a substantial increase in the morphological information and phylogenetic context for the base of Archosauromorpha.

I have made quite a few small comments throughout. Most of these are small comments on spelling or phrasing. I have noted a few different opinions I have on codings, but I think these should simply be addressed in some way in the text rather than a full-on recoding and rerunning of the extensive analyses. In some of the character descriptions, there are few to no examples of one or both states. I have noted where these severely limited my understanding of the morphology being described.

I congratulate the author on producing this extensive work, which will definitely contribute to my own future systematic studies and those of anyone studying the early diversification of archosaur-line reptiles.

·

Basic reporting

Firstly, I should say that this is by no means as thorough a review as I would like - it is a gigantic manuscript that deserves a much more careful review that I was able to give it.

On receiving the manuscript and seeing the full extent of the task, I was even less favorably disposed towards it when I noticed on the first page of the introduction that the author failed to spell their own name correctly! This speaks to the potentially very significant task of cleaning the manuscript up for publication. Nevertheless, having said that, in general I think there is a great deal of merit to the manuscript. It covers a lot of ground and the figures are largely very instructive (although a bit fuzzy in one or two cases – although I concede that the quality of the fossils is perhaps a major contributory factor here)! It is particularly nice to see some serious consideration given to forms such as Jesairosaurus, Prolacertoides and Kadimakara. Given my complaint about the length of the manuscript I hesitate to say that a little more detailed assessment of certain other basal archosauromoprhs would have been a great service to the scientific community, but that can certainly be taken on in further manuscripts!

The manuscript covers a lot of ground concerning Permo-Triassic archosauromorphs and it seems to be a very comprehensive study – particularly with respect to the proterosuchians. I think the author is to be applauded for his in-depth studies. I noted a number of very significant observations on various taxa and I believe that the author has been very diligent and consistent in his first-hand observations of material. I just hope he has been as diligent in his writing and particularly the bibliography!

Nothing of major significance jumped out at me that I would like to see the author address. Therefore with the caveat that I have not checked the references, re-run the analyses or carefully cross-referenced all the observations, I suggest that the manuscript will only need relatively minor revision. In other words a good copy-edit is needed but perhaps not too much more!

Experimental design

Very standard approach to the phylogeny and I have no complaints. However, in view of the length of the manuscript it would have been impossible for me to run the analyses myself without severely delaying the review. So reluctantly, I cannot confirm the accuracy of this aspect of the manuscript.

Validity of the findings

From my experience of some of these fossils I have no reason to believe that there is anything inaccurate in the observations. As noted above I cannot comment on the analyses, but nothing in the resulting cladograms gives me cause to question the accuracy of the analyses. I did have a look at a handful of the character scorings and I saw nothing amiss.

Additional comments

The very detailed descriptions and the well-labeled figures are extremely informative. This is a tremendous body of novel work and it will be an invaluable reference.

I would have appreciated some advance warning so that I could have been a bit more thorough in my review!

---

## Round 0.2 · Minor Revisions

· Academic Editor

Minor Revisions

Dear Martin,

Because PeerJ does not provide copy-editing and this is such an important paper, I took it on myself to edit your manuscript for language and style. I will e-mail you the Word document with the track changes as PeerJ only allows uploading PDF copies. Please check the track changes and, once you are are satisfied, resubmit the manuscript to PeerJ.

Many thanks.

Best, Hans

---

## Round 0.3 · accepted · Accept

· Academic Editor

Accept

Thank you for reviewing the edited version of your revised manuscript. I will now recommend the revised submission for acceptance for publication.